Vascular plant biodiversity of the lower Coppermine River valley and vicinity (Nunavut, Canada): an annotated checklist of an Arctic flora

http://orcid.org/0000-0003-1790-4332 Saarela Jeffery M. jsaarela@mus-nature.ca
http://orcid.org/0000-0002-7053-8557 Sokoloff Paul C.
Bull Roger D.
Botany Section and Centre for Arctic Knowledge & Exploration, Research and Collections, Canadian Museum of Nature , Ottawa, ON , Canada
Cowling Richard
Electronic publication date: 2017 Jan 31
Publication date: 2017
Volume: 5
Electronic Location ID: e2835
Received 2016 Sep 2; Accepted 2016 Nov 28
Copyright: © 2017 Saarela et al.
Copyright year: 2017
Copyright holder: Saarela et al.
License: This is an open access article distributed under the terms of the Creative Commons Attribution License, which permits unrestricted use, distribution, reproduction and adaptation in any medium and for any purpose provided that it is properly attributed. For attribution, the original author(s), title, publication source (PeerJ) and either DOI or URL of the article must be cited.
License URL: https://creativecommons.org/licenses/by/4.0/

Keywords: Floristics, Flora, Bloody Falls, Kugluktuk, Range extension, Subarctic, Herbarium specimens

Funding: Polar Continental Shelf Program project 50514 Canadian Museum of Nature Research support was provided by the Polar Continental Shelf Program (project 50514) and the Canadian Museum of Nature. The funders had no role in study design, data collection and analysis, decision to publish or preparation of the manuscript.

==============================
The Coppermine River in western Nunavut is one of Canada’s great Arctic rivers, yet its vascular plant flora is poorly known. Here, we report the results of a floristic inventory of the lower Coppermine River valley and vicinity, including Kugluk (Bloody Falls) Territorial Park and the hamlet of Kugluktuk. The study area is approximately 1,200 km2, extending from the forest-tundra south of the treeline to the Arctic coast. Vascular plant floristic data are based on a review of all previous collections from the area and more than 1,200 new collections made in 2014. Results are presented in an annotated checklist, including citation of all specimens examined, comments on taxonomy and distribution, and photographs for a subset of taxa. The vascular plant flora comprises 300 species (311 taxa), a 36.6% increase from the 190 species documented by previous collections made in the area over the last century, and is considerably more diverse than other local floras on mainland Nunavut. We document 207 taxa for Kugluk (Bloody Falls) Territorial Park, an important protected area for plants on mainland Nunavut. A total of 190 taxa are newly recorded for the study area. Of these, 14 taxa (13 species and one additional variety) are newly recorded for Nunavut (Allium schoenoprasum, Carex capitata, Draba lonchocarpa, Eremogone capillaris subsp. capillaris, Sabulina elegans, Eleocharis quinqueflora, Epilobium cf. anagallidifolium, Botrychium neolunaria, Botrychium tunux, Festuca altaica, Polygonum aviculare, Salix ovalifolia var. arctolitoralis, Salix ovalifolia var. ovalifolia and Stuckenia pectinata), seven species are newly recorded for mainland Nunavut (Carex gynocrates, Carex livida, Cryptogramma stelleri, Draba simmonsii, Festuca viviparoidea subsp. viviparoidea, Juncus alpinoarticulatus subsp. americanus and Salix pseudomyrsinites) and 56 range extensions are reported. The psbA-trnH and rbcL DNA sequence data were used to help identify the three Botrychium taxa recorded in the study area. Three new combinations are proposed: Petasites frigidus subsp. sagittatus (Banks ex Pursh) Saarela, Carex petricosa subsp. misandroides (Fernald) Saarela and Carex simpliciuscula subsp. subholarctica (T. V. Egorova) Saarela.

Introduction

Botanical exploration of the Canadian Arctic, encompassing the Canadian Arctic Archipelago and portions of mainland Yukon, Northwest Territories, Nunavut, Quebec (Nunavik) and Newfoundland and Labrador (Nunatsiavut), has been ongoing for nearly 200 years (see Saarela et al., 2013a for a brief summary). Mainland Northwest Territories and Nunavut include both boreal and Arctic ecosystems and the transitional ecotone between them (the forest-tundra or treeline), with a rich vascular plant flora comprising over 800 species (Porsild & Cody, 1980). A detailed summary of botanical explorations of continental Northwest Territories and Nunavut up to 1976 is given in Porsild & Cody (1980). Their classic flora, now over 35 years old, is the only comprehensive work for this large region. Many specimens and a considerable amount of new information on the Arctic flora of the continental Northwest Territories and Nunavut have accumulated since its publication, including the results of floristic inventories and ecological studies (Cody, Scotter & Zoltai, 1984, 1989, 1992; Gould & Walker, 1997; Saarela et al., 2013a). Further range extensions, additions to the flora of the region and other new information have been published in a series of papers (Cody, 1996a; Cody et al., 2000; Cody & Johnston, 2003; Cody, Reading & Line, 2003; Cody & Reading, 2005). Nevertheless, many Arctic areas of mainland Canada are unexplored or underexplored botanically. One such area is the lower Coppermine River valley in western Nunavut.

Floristics is the study of the distribution, number, types and relationships of plant species in an area. Here, we report the results of a floristic study of the lower Coppermine River valley and vicinity, in the form of an annotated checklist: a verified list of all vascular plant species in a defined area (Funk et al., 2007). We synthesise the results of fieldwork in 2014 and existing published and unpublished information, providing comprehensive baseline biodiversity data on the vascular plants of the area. Checklists have many uses: they can contribute to biogeographical, evolutionary and phytogeographical studies, provide information relevant to conservation efforts, provide data for comparative biodiversity analyses, help identify areas that harbour rare species, and help with the identification of organisms. They serve as foundational information for further floristic or ecological studies of a region, such as detailed characterisation of vegetation or setting up plots for long term vegetation monitoring. Checklists can also be useful in monitoring the effects of climate change on the composition of the flora by documenting if, where and when a particular species has been collected. Specimens that underpin floristic research are useful for research in systematics and taxonomy, phylogeography and DNA barcoding.

Coppermine River

The Coppermine River is one of Canada’s great Arctic rivers. It flows north 845 km from its headwaters at Lac de Gras, Northwest Territories (64°30′N, 110°30′W; approximately 300 km north of Yellowknife) through the forest-tundra and the Arctic shrub tundra to Coronation Gulf on the Arctic Ocean at the hamlet of Kugluktuk in the Kitikmeot Region of Nunavut, dropping over 400 m in elevation along the way (Fig. 1). Humans have inhabited the lower Coppermine River area for more than 3,000 years, beginning with Pre-Dorset people (McGhee, 1970). The river’s traditional name is Kogloktok, which means “the place of moving water” in Inuktitut, the Inuit language. In Inuinnaqtun, the primary Inuktitut dialect in the area, the river is known as Qurluktuk. The Coppermine River cuts through rock and sediment, resulting in a series of narrow gorges that produce rapids and wider valleys where sediment deposits have been washed away. The mouth of the river at Kugluktuk comprises a series of islands and shoals formed by sediments carried down the river, forming a large delta (Dredge, 2001). Today the Coppermine River is a well-known and popular wilderness canoe route. The 450 km Nunavut portion of the Coppermine River was nominated for Heritage River status under the Canadian Heritage River System in 2002.

Figure 1 Map of Canada showing the location of the lower Coppermine River and vicinity (the study area) in western Nunavut.

Red lines delimit the Canadian Arctic Ecozone, according to the Circumpolar Arctic Vegetation Map (CAVM Team, 2003). Map data © Google.

The Coppermine River area is important in the history of Arctic exploration. The river was given its English name by Samuel Hearne, an employee of the Hudson’s Bay Company who was the first western explorer to visit it. Hearne travelled to the river by foot in 1771 from Fort Prince of Wales, Hudson Bay, in search of the source of copper used by Inuit inhabiting the area (Hearne, 1795). Some 70 years later, the Coppermine River was explored as part of Sir John Franklin’s first (1819–1822) and second (1825–1827) overland expeditions to the northern coast of North America. Several of the numerous rapids along the lower Coppermine River bear names given to them during Franklin’s first expedition, including Escape Rapids (67°37′0″N, 115°28′0″W), Sandstone Rapids (67°27′0″N, 115°38′15″W), Muskox Rapids (67°22′55″N, 115°43′15″W) and Rocky Defile Rapids (67°2′0″N, 116°12′0″W) (Fig. 2). These are important landmarks in the area and are referred to in our collection locality descriptions, when appropriate. Sketches of the Coppermine River area made by Franklin expedition members Robert Hood and George Back, and published in Franklin (1823), are among the earliest visual records of the Canadian Arctic. However, some of these do not accurately illustrate the landscape, as demonstrated by St-Onge (1982), who compared some of the illustrations with contemporary photographs of the same places.

Figure 2 Rapids on Coppermine River.

(A) Sandstone Rapids along the Coppermine River. The upper slopes on the east side of the river are dominated by white spruce. (B) Waterfall flowing into the Coppermine River at the start of Escape Rapids, looking east. (C) Start of Muskox Rapids looking north along the Coppermine River. Photographs by P. C. Sokoloff (A, B) and R. D. Bull (C).

The next explorers to the Coppermine River were Peter Warren Dease and George Simpson, who visited the area during 1837–1839 as part of an assignment to map the Arctic coast from the mouth of the Mackenzie River eastward to the Kent Peninsula (Sampson, 1976; Barr, 2002). Later early explorers included David Hanbury in 1901 (Hanbury, 1904), George Douglas in 1911–1912 (Douglas, 1914) and the members of southern team of the Canadian Arctic Expedition in 1915 (Jenness, 2011), among others (Hodgins & Hoyle, 1997). All of the published accounts of these early explorers include important geographical information on the area. A trading post was set up in 1916 at the mouth of the Coppermine River, and the settlement of Coppermine was established in 1927, when the Hudson’s Bay Company moved its trading post there from Bernard Harbour northwest of the settlement. The community is situated on a rocky hill that slopes gradually north to the shore of Coronation Gulf. The name of the community was changed to Kugluktuk on 1 January 1996. We refer to the community as Kugluktuk even when referencing events before 1996.

The Coppermine River is located on the northwestern portion of the Canadian Shield. The geology and landforms of the lower Coppermine River valley and Kugluktuk are described by Dredge (2001) and the sedimentary sequence of the area is described by St-Onge (2012). The entire Coppermine River was covered by the Laurentide Ice sheet during the most recent glaciation (Ritchie, 1987). The Coppermine River valley south of Rocky Defile Rapids was freed from ice by 10,250 14C years before present (St-Onge, 1980), and the river valley to Coronation Gulf was ice-free by 10,000 years before present (Bruneau, 1985). In late-glacial times a portion of the southern half of the Coppermine River valley, south of the Coppermine Mountains, was covered by a large lake called Glacial Lake Coppermine (St-Onge, 1980). Lower lying areas along the river comprise marine deltaic boulder gravel, originating from a delta of the postglacial sea.

Bloody Falls

A total of 13 km south of the mouth of the Coppermine River (at the Arctic coast) is Bloody Falls, the first major rapid (Fig. 3). Although Bloody Fall (singular) is the official name of the geographical feature (Canadian Geographical Names, 2015), we here use Bloody Falls (plural) as is common. The Bloody Falls area is also known as Kugluk, which means rapids in Inuinnaqtun (Cameron, 2015). It is an important fishing spot, and archaeological investigations show it has been used by humans for over 3,000 years (McGhee, 1970). Bloody Falls is a series of rapids named by Samuel Hearne who, on 17 July 1771, witnessed his Chipewyan companions massacre a group of Inuit camped in the area (Hearne, 1795). The location of a tent ring site on a hill overlooking Bloody Falls is thought to be the area where the massacre occurred (Morrison, 1981), and this incident has received considerable study (e.g., Brand, 1992; Cameron, 2015). The fast-moving rapids of Bloody Falls are the result of a narrow and rapid descent of the river. The gorge through which the Coppermine River flows, forming the rapids, cuts through a bedrock sill (gabbro) (Fig. 4A), and the river drops about 10 m from the start to the end of the falls (Dredge, 2001). Above and below the rapids, the river is substantially broader, having eroded large glaciofluvial deposits beyond the northern and southern limits of the gabbro sill (Fig. 4B).

Figure 3 Map showing the extent of our study area along the lower Coppermine River and vicinity.

Orange circles denote 2014 collecting sites. Limits of the Fockler Creek area and Kugluktuk study sites are indicated with white squares. The inset shows the boundaries of Kugluk (Bloody Falls) Territorial Park. The red line delimits the Canadian Arctic Ecozone, according to the Circumpolar Arctic Vegetation Map (CAVM Team, 2003). All collections from Fockler Creek and sites to the south are part of the Subarctic forest-tundra. Map data © Google, IBCAO, Landsat.

Figure 4 Kugluk (Bloody Falls) Territorial Park.

(A) Rocky beach just above the start of Bloody Falls on the Coppermine River in Kugluk (Bloody Falls) Territorial Park. The fast-moving rapids of Bloody Falls are the result of a narrow and rapid descent of the river that starts here. The gorge through which the Coppermine River flows, forming the rapids, was cut through a bedrock sill (gabbro), demarcated in this photo by the vertical cliffs. The upland tundra on the east side of the Coppermine River (top of photo) is not part of Kugluk (Bloody Falls) Territorial Park. (B) Shrub tundra and sand dunes along Coppermine River valley in Kugluk (Bloody Falls) Territorial Park, looking northeast. The broad Coppermine River in the centre of the photo is just below Bloody Falls. (C) A braided stream at the bottom of a deep gully in Kugluk (Bloody Falls) Territorial Park. (D) One of two small ponds just west of Bloody Falls in Kugluk (Bloody Falls) Territorial Park. Photographs by R. D. Bull (A), P. C. Sokoloff (B), and J. M. Saarela (C, D).

In recognition of the long history of pre-contact use of Bloody Falls, the area was designated as Bloody Falls National Historic Site of Canada in 1978, because “it was a traditional fishing site, also containing small caribou hunting stations, that record the presence of Pre-Dorset, Thule, First Nation and Inuit cultures over the past three millennia” (Parks Canada, 2015). More recently, in 1995 a 10.5 km2 area around Bloody Falls was designated by the then Government of Northwest Territories as Kugluk (Bloody Falls) Territorial Park (Fig. 3) (Nunavut Parks & Special Places, 2008). The park is located on both sides of the Coppermine River, centred on Bloody Falls. On the west side of the river, the park borders the river. On the east side, a small portion of the park above the start of Bloody Falls borders the river and the remainder is separated from the river by a parcel of Inuit Owned Land (Fig. 3). Development in the park includes picnic tables, fire pits, an outhouse, informational signage, a boardwalk, and an all terrain vehicle (ATV) trail. The history of designating the Bloody Falls area as a territorial park is discussed by Cameron (2015).

The immediate area around the base of Bloody Falls was described by McGhee (1970), as part of his archaeological research. Just north of what is currently a day-use area is an older river course comprising two small ponds bordered by steep rocky slopes (Fig. 4D). Between this canyon and the one through which the Coppermine River runs is a basalt remnant reaching 20 m above the river. The eastern and southern borders of this remnant along the river are near vertical cliffs and ledges, whereas to the north it gradually slopes to a flat, seasonally flooded rocky beach. The apex of the basalt is covered in thick willow thickets (Salix glauca L.). McGhee (1970) reported that willows had appeared there only in the previous decades, according to local people.

On the west side of the Coppermine River within the territorial park are numerous large and steep-sloped sandy gullies (Fig. 4C), some up to 80 m deep. These were formed by erosion of sandy postglacial deltas, a process that is ongoing (Dredge, 2001). Active dunes are present along the tops of some of the gullies. Some of the gullies have streams, often braided ones, flowing through their bases (Fig. 4C), while the bases of others are dry. Some of the gullies provide protected habitat where lush willow thickets thrive. In addition to the dramatic Bloody Falls rapids, these large gullies are an attractive and unique feature of Kugluk (Bloody Falls) Territorial Park. There are no such gullies elsewhere in the study area, to our knowledge.

Ecology

Our study area along the lower Coppermine River valley and vicinity extends from the Bigtree River (66°56′23.8″N, 116°21′3.2″W) some 112 km southwest of Kugluktuk and north along the river to Kugluktuk at the Arctic coast (Fig. 3). It includes islands in the mouth of the Coppermine River and a region extending some 20 km northwest of Kugluktuk to Richardson Bay at the confluence of the Rae and Richardson rivers at Coronation Gulf. The study area is approximately 1,200 km2, and comprises Arctic and Subarctic regions (Fig. 3).

The southern limit of the Arctic in Canada is equivalent to the northern limit of the forest-tundra ecozone, as defined by the Circumpolar Arctic Vegetation Map (CAVM Team, 2003; Walker et al., 2005). They followed Timoney et al. (1992), who defined the boundary between the forest-tundra and Arctic ecozones by the ratio of trees to upland tundra. The southern limit of the Arctic along the Coppermine River has a tree:upland tundra cover ratio of 1:1,000 (Timoney et al., 1992). This boundary parallels an approximately three kilometres portion of the Coppermine River that runs east to west, approximately 27 km south-southwest of Bloody Falls and seven kilometres north of Sandstone Rapids (Fig. 3). The northern limit of the treeline or forest-tundra in Canada’s Arctic is not necessarily the northern limit of a tree species, as trees may grow well beyond it as scraggly bushes, low scrub or prostrate mats. Along the Coppermine River valley the Subarctic-Arctic boundary is more than 12 km south of the northern limit of white spruce (Picea glauca (Moench) Voss), the northern-most tree species in this part of the Canadian Arctic, whose relative abundance is used to define the ecozones.

The distribution of white spruce in this area was described by Geurts (1983) as part of a study of pollen assemblages along the Coppermine River valley. Continuous white spruce trees up to 8–10 m tall extend along the eastern side of the river to 67°19′N, about 1.5 km south of Muskox Rapids (Fig. 5A). Between 67°19′N and 67°39′N, white spruce is only present along valley edges (Figs. 5B and 5C), and north of Muskox Rapids the spruce forest becomes increasingly fragmented and the trees begin to take the form of Krummholz, quickly disappearing from the landscape altogether (Fig. 5D). The last trees observed by Geurts (1983) were at 67°39′N, 2.5 km north of Escape Rapids. The northernmost spruce stands may be relicts from warmer periods during the mid-Holocene (Nichols, 1975, 1976). There is evidence that spruce was distributed further north in the past. Nichols (1976) reported spruce macrofossils from the Kugluktuk area along the coast that dated to 3,715 ± 120 years before present, and Nichols (1975) noted spruce macrofossils from a site (Saddleback Hill) about one kilometre south of Kugluktuk, indicating spruce grew near the coast from 3,700 to 2,500 years before present, surviving in a vegetative state.

Figure 5 Coppermine River.

(A) Forest-tundra along the Coppermine River at the start of Rocky Defile Rapids. (B) South-facing slopes on W side of Coppermine River, about halfway between Escape Rapids and Muskox Rapids (ca. 67°31′18.2″N, 115°36′20.1″W). This is just north of the southern limit of the Arctic ecozone. Habitat comprises scattered white spruce (Picea glauca) and shrub tundra. Sparganium hyperboreum was collected in the pond in the foreground. (C) White spruce forest along Coppermine River ca. 7.8 km NNE of Sandstone Rapids, just north of the southern limit of the Arctic ecozone. (D) Tundra and clay mounds along east side of Coppermine River, south of Bloody Falls. Photographs by J. M. Saarela (A, D) and P. C. Sokoloff (B, C).

The Arctic has been divided into various phytogeographical and bioclimatic ecozones. Porsild & Cody (1980) divided the continental Northwest Territories (including what is now continental Nunavut) into five phytogeographical provinces. The study area includes portions of their phytogeographical provinces “4” and “5”. They defined “4” as including the treeless part of the [former] District of Mackenzie and most of the [former] District of Keewatin, and “5” as comprising the wooded (non-Arctic) portion of the region. In the classification of Canada’s National Ecological Framework, all of the study area is part of the southern Arctic ecozone and the Coronation Hills ecoregion, located between Amundsen and Coronation Gulfs and the northeast shore of Great Bear Lake. In the Coronation Hills ecoregion the mean annual temperature is −11 °C, the summer mean is 5 °C and the winter mean is −26 °C (Ecological Stratification Working Group, 1995). In Kugluktuk from 1981 to 2010, mean annual temperature was −10.3 °C, summer mean (June–August) temperature was 8.4 °C and winter mean temperature (December–February) was −26.5 °C (Environment Canada, 2015).

The CAVM (CAVM Team, 2003; Walker et al., 2005) classified the circumpolar Arctic into five bioclimatic subzones (A to E, with A being the coldest and harshest and E the warmest and least harsh), 23 floristic provinces and eight aboveground plant biomass classes based on normalised difference vegetation index (NDVI) data. They also described the general appearance of the vegetation based on dominant growth forms (dominant vegetation physiognomy) based on satellite imagery. The Arctic portion of the study area is part of bioclimatic subzone E, which is characterised by a mean July temperature of 9–12 °C, a summer warmth index of 20–35 °C, a dense vertical structure of plant cover (with 2–3 layers, moss layer 5–10 cm thick, herbaceous/dwarf shrub later 20–50 cm tall, and sometimes with a low-shrub layer to 80 cm), 80–100% plant cover, total phytomass of 50–100 t ha−1, net annual production of 3.3–4.3 t ha−1 yr−1, and 200–500 vascular plant species in local floras. Following other bioclimatic zonation approaches, the Arctic portion of the study area would be classified as low Arctic (Polunin, 1951; Bliss, 1997); low, erect shrub zone (Edlund, 1990); Arctic shrub (Daniëls et al., 2000) or low shrub (Walker et al., 2002). The Arctic portion of the study area is part of the poorly defined (Elvebakk, 1999) Central Canada floristic province, which extends from the Yukon/Northwest Territories border east to about Chantrey Inlet, Nunavut (67°16′00″N, 95°14′00″W) and north to Meighen Island, Graham Island and the western half of Devon Island, Nunavut. Physiognomically, the lower half of the study area that is part of the CAVM is classified as S2 low shrub tundra. To the north, the portion of the study area on the east side of the Coppermine River is classified as G3 (non-tussock sedge, dwarf-shrub moss tundra) and the west side of the Coppermine River is classified as G4 (tussock sedge, dwarf-shrub moss tundra).

Previous collecting in the study area

19th century collections

There have been no previous attempts at a comprehensive floristic survey of the Coppermine River valley or Kugluktuk, but numerous plant collections have been made in the region. The first collections were made in 1821 by Sir John Richardson, surgeon and naturalist on Franklin’s first overland expedition. These collections were published in an appendix to Franklin’s narrative of that trip (Richardson, 1823). The appendix was also issued separately, with addenda by Robert Brown (Richardson & Brown, 1823). Franklin’s party descended the Coppermine River and travelled north from Point Lake, Northwest Territories (65°14′N, 113°09′W), to the Arctic coast. Most of the Richardson’s plant collections were abandoned (due to running out of supplies after the party reached the coast and the onset of winter), except for some from along the Coppermine River. These were carried back to Point Lake by W. W. F. Wentzel, who along with four other men left Richardson and the rest of the main party upon their arrival at the mouth of the Coppermine River (Richardson, 1823; Houston, 1984). The surviving plant collections are indicated by “(B)” in Richardson’s botanical appendix, denoting the “Barren grounds” region from Point Lake to the Arctic Sea (Richardson, 1823: 2). He recorded some 117 vascular plant taxa from this region. We have not attempted to fully align the nearly 200 year-old nomenclature and species concepts used by Richardson with current ones, where they differ.

Richardson (1823) described several new taxa from the “Barren grounds”, some of which are currently accepted. These include Arenaria propinqua Richardson (=Sabulina rubella (Wahlenb.) Dillenb. & Kadereit), Anemone borealis Richardson (=Anemone parviflora Michx.), Aster montanus Richardson (=Eurybia sibirica (L.) G. L. Nesom), Braya glabella Richardson, Cardamine digitata Richardson, Cineraria frigida Richardson (=Tephroseris frigida (Richardson) Holub), Chrysanthemum integrifolium Richardson (=Hulteniella integrifolia (Richardson) Tzvelev), Crepis nana Richardson (=Askellia pygmaea (Ledeb.) Sennikov), Hedysarum mackenzii Richardson (=Hedysarum boreale subsp. mackenziei (Richardson) S. L. Welsh), Ranunculus arcticus Richardson, Ranunculus purshii Richardson (=Ranunculus gmelinii DC.), Salix desertorum Richardson (=Salix glauca L.), Senecio lugens Richardson and Stellaria laeta Richardson (=Stellaria longipes Goldie). The protologues of only three of these contain locality descriptions more precise than area “B” denoting the provenances of the original material. Senecio lugens was described from Bloody Falls, and Braya glabella and Chrysanthemum integrifolium were described from the Copper Mountains [Coppermine Mountains].

Three additional new species collected in the “Barren grounds” were published in Richardson (1823) based on descriptions provided by Robert Brown. Traditionally the authority for these taxa has been attributed to “R. Br. ex Richardson”: Calamagrostis purpurascens R. Br. ex Richardson (collected in the barren grounds), Carex concinna R. Br. ex Richardson (collected in the barren grounds and “wooded country from latitude 54° to 64° north”) and Carex podocarpa R. Br. ex Richardson (collected in the barren grounds). However, since the specific names and their validating description were explicitly ascribed to Brown in Richardson (1823) by the statement “Brown, M. S.”, attribution to Brown is all that is required as the nomenclatural authority for these names (McNeill et al., 2012). Richardson (1823) also described other currently recognised Arctic taxa, based on collections from other areas, including Phlox hoodii Richardson, Tofieldia coccinea Richardson and Woodsia glabella R. Br.

The first sets of Richardson’s collections are housed at the British Museum (BM) and Royal Botanical Gardens Kew (K), with duplicate material at the National Herbarium of Canada (CAN), Field Museum of Natural History (F), Gray Herbarium (GH), The New York Botanical Garden (NY), Muséum national d’histoire naturelle (P), Philadelphia Academy of Natural Sciences (PH), and probably other herbaria. Information on the labels of the specimens is vague, with no dates or collection numbers and imprecise geographical descriptions such as “Arctic Coast”, “N. W. America”, “Arctic Plants”, “Arctic America” and “Sea coast”, and sometimes with only “Dr. Richardson”. This is not surprising, as part of the goal of the expedition was to map the north coast, which was unknown. Because of the lack of information on the specimens and in the protologues, identifying type or original material of Richardson’s new taxa has often been problematic or impossible, with many specimens annotated as “possible type or isotype” or similar statements. In most cases it is not possible to determine which specimens are unique collections (from different places at different times) and which are duplicates (from the same place and time).

In 1826 as part of Franklin’s second overland expedition (1825–1827) Richardson again travelled along the Coppermine River, this time south from the Arctic coast to the east end of Great Bear Lake (Franklin, 1828). We are not aware of any collections made from the Coppermine River area during this trip, but there are collections from other areas visited, such as the Mackenzie River. Richardson’s collections from the first two Franklin expeditions were studied by W. J. Hooker for his Flora boreali-americana (Hooker, 1840). Richardson travelled along the Coppermine River a third time, in the late 1840s, as part of an expedition in search of Franklin and his crew, who went missing in the Canadian Arctic in 1846. There is indication in Richardson’s account of that trip that he collected plants along the Coppermine River and/or in the areas visited before reaching the Coppermine River. He wrote on 11 September 1846 that along the Kendall River west of its confluence with the Coppermine River, “…I deposited my packet of dried plants and some books in a tree, intending to send for them in the winter” (Richardson, 1851). The specimens may have been retrieved, as some collections labelled “Arctic Sea Coast”, which may have been part of that packet, were referred to in a letter by Francis Boott to Richardson providing identification of Carex species collected on this expedition (Richardson, 1851).

20th and 21st century collections

Plant collections were apparently not made in the lower Coppermine River area again until 1915. Frits Johansen and Rudolf M. Anderson, members of the Southern Party of the Canadian Arctic Expedition 1913–1918 (Jenness, 2011), made a few winter collections along the Coppermine River in February 1915, which were reported in Macoun & Holm (1921). Throughout the rest of the 20th century, numerous collections were made in the area, mostly in Kugluktuk. Most of these were made incidental to other activities and few were made by professional botanists. Collection information for only a small subset of these collections has been published, either in floristic papers (Materials and Methods) or taxon-specific taxonomic treatments. The dots in the vicinities of Kugluktuk and Bloody Falls on the maps in Porsild & Cody (1980) are based on most of these earlier collections. Pioneering Canadian bush pilot Arthur M. Berry made collections in Kugluktuk in 1931 (specimens at CAN), while working for Northern Aerial Mineral Exploration (Piper, 2010); these were reported in Porsild (1943). Father Arthéme Dutilly of the Catholic University of America made collections in Kugluktuk in 1934 (CAN), during a three-month Arctic expedition to carry out botanical and entomological research (Anonymous, 1935; Boivin, 1983). L. Ross (affiliation unknown) made collections in Kugluktuk in 1940 (CAN). Hansford T. Shacklette, later of the U.S. Geological Survey, collected in Kugluktuk in 1948 (CAN) as part of an expedition that visited Port Radium, Sawmill Bay and Dease Arm of Great Bear Lake (Porsild & Cody, 1980). Mare Hammer of Denmark reportedly made collections from the Kugluktuk area in 1948 (Anonymous, 1949); we have neither seen any of these nor are we certain they exist. Alfred E. Porsild of the National Museums of Canada—the only professional botanist who collected in the area in the 20th century—made collections in Kugluktuk on 26 July 1949 (CAN). A stop in Kugluktuk is not recorded in Porsild’s account of his summer 1949 field season on adjacent Banks Island and Victoria Island, though he does report leaving Port Radium on 25 July 1949 and arriving at Holman Island Post [Ulukhaktok] on 28 July 1949 (Porsild, 1950a). Kugluktuk is about halfway between those two settlements and the party likely made a brief stopover there, during which Porsild would have explored the flora. R. E. Miller (affiliation unknown) made collections in Kugluktuk and at the mouth of the Rae River in July and August 1955 (CAN). Raymond D. Wood and Mildred Wood made collections in Kugluktuk in 1958. Wood, a lawyer by profession, was an avid photographer of plants, he and Mildred (“the botanist of the team” according to his obituary) collected vouchers of the plants they photographed, although the vouchers list only Raymond’s name as collector. Their collections from Kugluktuk at CAN are associated with colour slides that are part of a larger set of their slides housed in the archives of the Canadian Museum of Nature (Porsild, 1965). James A. Larsen, ecologist from the University of Wisconsin, made collections in Kugluktuk in 1962 (CAN). F. Fodor made collections in Kugluktuk in 1972 (UBC). Biologists Frederick W. Schueler and J. D. Rising made a collection in Kugluktuk in 1975 (CAN). Teresa Dolman made collections in Kugluktuk on 2 August 1995 (LEA). Carolyn Parker and I. Jonsdottir made a few collections (ALA) in Kugluktuk on 23 June 1999. Lawrence K. Benjamin made collections in Kugluktuk in June and July 2000 (ACAD).

The largest collection of plant specimens from the study area was gathered in 1951 at Kugluktuk and Bloody Falls by W. I. Findlay, who was then a student assistant with the Canada Department of Agriculture (Porsild & Cody, 1980). The first and most complete set of these specimens is deposited in the National Collection of Vascular Plants (DAO), Agriculture and Agri-Food Canada, Ottawa, and duplicates are housed in ACAD, ALTA, LEA, MT, QFA, UBC and possibly elsewhere. These collections were published by Cody (1954b). This is by far the most comprehensive historical collection of material from the area, but it did not represent a complete survey of the flora, given the lack of specimens of some widespread, common taxa. For example, no species of Poa and only three species of Carex, the most species-rich genus of plants in the Arctic, were collected. Findlay also made collections at the mouth of the Napaaktoktok River (67°49′N, 114°44′), just east of the current study area.

A few collections have been made in the study region in the 21st century. Jonathan D. Davis made some plants collections (CAN, GH, MACF, and OAC) in 2006, as part of an ethnobotanical study of the Kiluhikturmiut Inuinnait of Kugluktuk (Davis & Banack, 2012). They found that eight species from six families, and one plant-derived substance, are part of the traditional diet; six types of plants from three families are used as traditional raw materials; and eleven species are used as medicine. Our team in 2008 (L. J. Gillespie, J. M. Saarela, L. M. Consaul & R. D. Bull) made five collections (CAN) outside the Kugluktuk Airport while waiting for a flight to Cambridge Bay. Bruce Bennett made 17 collections (ALA, BABY, CAN, and UBC) around the Kugluktuk Airport during a brief stopover there on 21 July 2013.

Materials and Methods

In July 2014, we explored and collected plants along the lower Coppermine River and vicinity. Fieldwork was conducted under the following licenses or permissions: Nunavut Department of Environment Wildlife Research Permit 2014-034; Nunavut Territorial Parks Use Permit 2014-01; an Inuit Owned Land Exemption Certificate from the Kitikmeot Inuit Association Department of Land, Environment and Resources to access Inuit Owned Land in the Kitikmeot Region of Nunavut; and Approval for the Use of Waters or Deposit of Waste Without a Licence (Approval number 8WLC-FCA1415) from the Nunavut Water Board.

We established three base camps, one each at Fockler Creek (campsite coordinates: 67°25′49″N, 115°37′54″W), Kugluk (Bloody Falls) Territorial Park (67°44′34″N, 115°22′22″W), and Kugluktuk (67°49′29″N, 115°4′57″W) (Figs. 3 and 6). We spent 10 days at Fockler Creek (1–10 July 2014) exploring an area of approximately 10 km2, 11 days at Kugluk (Bloody Falls) Territorial Park (11–22 July 2014), and nine days in Kugluktuk (29–30 June 2014 and 22–29 July 2014), an area of ca. 3 km2 excluding sites west of town in the vicinity of Heart Lake. The Fockler Creek camp was south of the limit of the Arctic ecozone, while the other two camps were within the Arctic ecozone (Fig. 3) (CAVM Team, 2003). Fockler Creek is a small river that runs into the east side of the Coppermine River just south of Sandstone Rapids (Figs. 6A and 7). Our collection sites in the vicinity of Fockler Creek are described in reference to their distances from Sandstone Rapids (67°27′00″N, 115°38′15″W). Other named geographical features in the Fockler Creek area are Sleigh Creek and Tundra Lake. Sleigh Creek is a small river that parallels Fockler Creek to the north and flows northwest from the narrow Tundra Lake.

Figure 6 Maps showing the locations of three main study areas.

(A) Fockler Creek area. (B) Kugluk (Bloody Falls) Territorial Park and vicinity (sites on the east side of the Coppermine River are outside the park boundary). (C) Kugluktuk. Orange circles denote 2014 collecting sites. Circle size is correlated with the number of collections per site (see legend). Map data © Google, SIO, NOAA, U.S. Navy, NGA, GEBCO, Landsat.

Figure 7 Fockler Creek.

(A) White spruce forest and rocky floodplain in the valley along Fockler Creek (ca. 67°25′49.50″N, 115°37′41.27″W), looking northwest. (B) White spruce forest-tundra along Fockler Creek, looking northwest. (C) Shrub tundra and white spruce forest in the valley on the north side of Fockler Creek. (D) Meadow along a small unnamed creek just south of Fockler Creek, looking north. A small stand of balsam poplar (Populus balsamifera) is present to the right of the white spruce stand on the south-facing slope above the creek. Photographs by P. C. Sokoloff (A), R. D Bull (B), and J. M. Saarela (C, D).

Our aim was to document all of the vascular plant species in the vicinity of each of our three camps with at least one voucher collection; for many species we made multiple collections per area. We visited and explored as many habitats as possible, by foot, at each camp, and made collections as we encountered taxa. We also explored several more remote areas by helicopter over two days (7–8 July 2014), staging from Fockler Creek (Fig. 3). By helicopter we visited five forest-tundra sites (all south of the Arctic ecozone) along the Coppermine River south of Fockler Creek. These included the forest-tundra adjacent to the mouth of Bigtree River (66°56′5″N, 116°20′10″W) on the west side of the Coppermine River (Fig. 8); the mouth of the Kendall River (67°7′0″N, 116°7′0″W), a large river that drains the Dismal Lakes into the Coppermine River; the mouth of Melville Creek (67°15′40″N, 115°31′20″W), a river draining lakes on the east side of the Coppermine River (Fig. 9); the Coppermine Mountains (67°18′N, 116°0′W), which rise 400–600 m to the north and west of the Coppermine River (Fig. 10); and a large esker (67°22′40″N, 115°42′38.5″W) on the east side of the Coppermine River (see Fig. 13 in St-Onge, 1980 for esker details), approximately 0.6 km south-southeast of Muskox Rapids. Helicopter sites along the Coppermine River within the Arctic ecozone included a site on the north side of the Coppermine River (67°31′18.2″N, 115°36′20.1″W), approximately halfway between Escape Rapids and Muskox Rapids (this site was just above the southern Arctic limit) (Fig. 5B); a site just above (south of) Escape Rapids (67°37′N, 115°28′W) on the west side of the Coppermine River; and an unnamed island in the mouth of the Coppermine River approximately 3.3 km east of Kugluktuk (67°49′29″N, 115°1′3.2″W) (Fig. 11C). Our two other helicopter stops were at a coastal site (Richardson Bay) near the confluence of the Rae and Richardson rivers at Coronation Gulf (67°54′11.2″N, 115°32′27.4″W) (Fig. 12) and the peninsula along the northwest side of Expeditor Cove (67°52′44″N, 115°16′58″W). Because search time at helicopter sites was limited, we were unable to exhaustively inventory them. Rather, we focused on making collections of species that were rare at the site or in the study area, but also made collections of common or interesting species as time permitted. At Kugluktuk, we explored numerous areas within the community and adjacent natural areas (Fig. 11). We also explored the Heart Lake area, approximately 7.5 km southwest of the mouth of the Coppermine River at Kugluktuk. We travelled to this area by ATV and explored the area while walking back to Kugluktuk.

Figure 8 Big Creek.

(A) Forest-tundra along Big Creek. Dominant species include Picea glauca, Salix glauca, Rhododendron tomentosum subsp. decumbens (white flowers) and Dasiphora fruticosa (yellow flowers). (B) Forest-tundra on slopes above Big Creek. Dominant species include Picea glauca, Salix glauca, Hedysarum americanum, and Rhododendron lapponicum. Photographs by J. M. Saarela.

Figure 9 Melville Creek.

(A) Just east of Melville Creek’s confluence with the Coppermine River. The rocky shore along the edge of the river transitions to shrub tundra with scattered white spruce, and upland areas on the north and south side (not shown) of the river are dominated by white spruce. (B) Scattered white spruce forest in the forest-tundra. Melville Creek is in the foreground, Coppermine River is in the background, looking north. Photographs by P. C. Sokoloff (A) and R. D. Bull (B).

Figure 10 Coppermine Mountains.

(A) The Coppermine Mountains on the west side of the Coppermine River, looking northeast. (B) View looking south from atop the Coppermine Mountains, overlooking the forest-tundra along the Coppermine River. Photographs by J. M. Saarela.

Figure 11 Kugluktuk.

(A) Shrub tundra and rocky outcrop in Kugluktuk, looking north towards Coronation Gulf. (B) Tundra in the valley in the southern part of Kugluktuk, between the power plant and the southwestern part of the community. (C) Beach on a small unnamed island just northeast of Kugluktuk at the mouth of the Coppermine River. (D) View of Kugluktuk facing east-southeast from the top of North Hill, a large gabbro sill in the northwest part of the community. Photographs by P. C. Sokoloff.

Figure 12 Rae River.

(A) Tundra just above the mouth of the Rae River at Coronation Gulf. (B) Low tundra flats at delta of Rae and Richardson rivers, facing south. Photographs by P. C. Sokoloff (A) and J. M. Saarela (B).

All of our collections were dried in the field in standard plant presses. For each collection we preserved a small sample of leaf tissue in silica gel for future molecular analyses. In most cases, we tagged the plant from which we obtained the sample. These tissue samples are housed at the Canadian Museum of Nature. The first set of our collections is deposited in the National Herbarium of Canada, Canadian Museum of Nature (CAN). Duplicates are deposited in the University of Alaska Museum of the North (ALA); the University of Alberta Vascular Plant Herbarium (ALTA); University of British Columbia Herbarium in the Beaty Biodiversity Museum (UBC); Botanical Museum in Oslo (O); Missouri Botanical Garden (MO); the Marie-Victorin Herbarium at the University of Montreal (MT); United States National Herbarium in the National Museum of Natural Sciences, Smithsonian Institution (US); and the University of Manitoba Herbarium (WIN). Locations of duplicate materials are indicated in the specimen citations.

We have attempted to account for all of the vascular plant collections that have been made in the region. All species reported for the study area are documented by one or more specimens and are summarised in an annotated checklist. To find specimens collected previously in the study area, we searched the collections at CAN and DAO, and we queried the Canadensys database (data.canadensys.net/explorer/en/search), which publishes biodiversity information from numerous Canadian herbaria. Through Canadensys we located relevant specimens housed in ALTA, MT, UBC, the E. C. Smith Herbarium at Acadia University (ACAD), the University of Lethbridge Herbarium (LEA), and Herbier Louis-Marie at Université Laval (QFA). We examined images of the specimens at ACAD and LEA, accessed through Canadensys. We have seen nearly all of the specimens mentioned in the text and cited under specimens examined, except those at MT and QFA, which are duplicates of Findlay collections. We saw only the specimens collected by B. Bennett housed at CAN; the others were determined by him. Images of many of John Richardson’s collections are available on the Kew website, but we did not critically examine these. Because of their lack of precise collection information, in most cases it is not possible to know whether or not the collections he cited from the “Barren lands” were made in the study area, or even, in some cases, the “Barren lands”. The coordinates for Kugluktuk listed on the labels of Findlay’s collections and in Cody (1954b) are 67°49′N, 115°5′W, a location in the middle of the mouth of the Coppermine River. In the specimen citations for these collections we report coordinates that correspond to central Kugluktuk, and estimate a one kilometre accuracy range. The labels for most specimens collected by others record the territory as Northwest Territories, the jurisdiction of which Nunavut was part until 1999. “Not recorded previously for Nunavut” or similar phrases in the text means not recorded for the portion of the former Northwest Territories that is now part of Nunavut. In the specimen citations, numbers immediately following accession numbers (i.e., numbers prefaced by a herbarium acronym) are barcode numbers.

Taxa in the annotated checklist are first organised by major clade: lycophytes, monilophytes, gymnosperms and angiosperms (monocots, eudicots), and families, genera and species are listed alphabetically under their respective higher-level taxon. We indicate in parentheses, the number of genera and the number of species in each family in the region. For example [1/2] indicates one genus and two species in the family. Numbers in parentheses after the number of species refer to the total number of taxa in the family recorded for the study area including additional infraspecific taxa. Family-level classifications follow Christenhusz, Zhang & Schneider (2011) for lycophytes, Christenhusz et al. (2011) for gynmnosperms, Smith et al. (2006) for monilophytes and Angiosperm Phylogeny Group III (2009) for angiosperms.

At the genus, species and infraspecific levels the nomenclature we use does not rely on a single source, but considers all of the available taxonomic information. We have extensively consulted relevant taxonomic and phylogenetic literature, as well as earlier (Porsild & Cody, 1980) and current floras, including available volumes of the Flora of North America series. An important source of information has been the Annotated Checklist of the Panarctic Flora (PAF): Vascular Plants (Elven et al., 2011), which evaluates the taxonomy and nomenclature of the global Arctic flora. In the many instances where treatments of species and/or species complexes differ among publications, we provide a brief discussion of the taxonomic issue(s) and indicate which treatment we follow. When accepted names differ among one or more of Elven et al. (2011), Porsild & Cody (1980), the Flora of North America treatment and other recent taxonomic work, we briefly discuss the names for the taxon applied in each treatment.

Common name(s) in English are derived mostly from the Flora of North America series and Brouillet et al. (2010+). Global distribution summaries are those given in Elven et al. (2011), and for each taxon we include a brief statement summarising the known distribution in the Canadian Arctic, referencing primary and secondary literature sources, and sometimes unpublished specimens. The primary literature sources are mostly those that document sites on mainland Nunavut additional to the ones mapped in Porsild & Cody (1980). For a subset of taxa we include photographs taken in the field to facilitate identification and illustrate ecology. For photographs with a corresponding voucher specimen, we provide collection numbers in the figure captions to allow cross-referencing of the photos and associated collection data. For photographs that do not have a corresponding voucher specimen, we include the locations and dates of the photographs in the figure captions.

New or existing collections are recorded as Noteworthy Records when they meet one or more of the following criteria: (1) they represent major or minor range extensions for a taxon, extending its known (i.e., published) distribution; (2) they are first record(s) for Nunavut; (3) they are first record(s) for mainland Nunavut; (4) they are first record(s) for the study area.

The electronic version of this article in portable document format (PDF) will represent a published work according to the International Code of Nomenclature for algae, fungi, and plants (ICN), and hence the new names contained in the electronic version are effectively published under that Code from the electronic edition alone. In addition, new names contained in this work which have been issued with identifiers by IPNI will eventually be made available to the Global Names Index. The IPNI LSIDs can be resolved and the associated information viewed through any standard web browser by appending the LSID contained in this publication to the prefix “http://ipni.org/”. The online version of this work is archived and available from the following digital repositories: PeerJ, PubMed Central, and CLOCKSS.

Results

We collected 1,380 numbers (Saarela nos. 3044–4424) representing 1,413 unique collections (a subset of collections was found to be mixed collections of more than one species). Of the 1,413 collections, two are cyanobacteria, 56 are bryophytes, nine are fungi, 107 are lichens, and 1,239 are vascular plants. Cyanobacteria and lichen collections are reported without interpretation in Appendix 1, and collection numbers for bryophytes and fungi are also listed there. Lichens were identified by Colin E. Freebury and R. Troy McMullin (Canadian Museum of Nature). We gathered sufficient material for 1–12 replicates (Fig. S1) of each vascular plant collection, representing at least 2,701 herbarium sheets (more if some of these are further subdivided). We examined some 240 unique collections previously collected in the study area, for a total of 1,653 unique collections from the area.

The vascular plant flora of the lower Coppermine River valley and vicinity includes 45 families, 134 genera, 300 species, eight infraspecific taxa, and three hybrids (311 taxa at species level or below) (Tables 1 and 2). Lycophytes are represented by one family. Ferns (monilophytes) are represented by six families, six genera, and 14 species. Gymnosperms are represented by two families, two genera, and two species. Monocots are represented by nine families, 31 genera, 100 species, and 104 taxa. Eudicots are represented by 27 families, 93 genera, 182 species, and 187 taxa. A complete listing of taxa is provided in Table 2 and in the annotated checklist. The number of collections for each of the taxa recorded ranges from 1 to 28 (mean 5.3 ± 3.9). A total of 48 taxa are known by a single collection, 42 by two, 27 by three, 31 by four, and 163 by five or more (Table 2). Taxa known from the greatest number of collections are Draba glabella Pursh (25 collections) and Salix glauca var. cordifolia (Pursh) Dorn (28 collections). Taxa known from one to three collections in the study area may be considered rare or uncommon.

Table 1 Number of genera and species in each major group of vascular plants in the lower Coppermine River valley and vicinity.

Taxon	Genera	Species	
Lycophytes (1)	
Lycopodiaceae	2	2	
Monilophytes (6)	
Cystopteridaceae	1	1	
Dryopteridaceae	1	1	
Equisetaceae	1	5	
Ophioglossaceae	1	3	
Pteridaceae	1	1	
Woodsiaceae	1	3	
Total	6	14	
Gymnosperms (2)	
Cupressaceae	1	1	
Pinaceae	1	1	
Total	2	2	
Monocots (9)	
Amaryllidaceae	1	1	
Cyperaceae	4	46 (1)	
Juncaceae	2	8	
Juncaginaceae	1	2	
Orchidaceae	2	2	
Poaceae	17	35 (4)	
Potamogetonaceae	2	4	
Tofieldiaceae	1	2	
Typhaceae	1	1	
Total	31	101 (106)	
Eudicots (27)	
Asteraceae	15	23	
Betulaceae	2	3	
Boraginaceae	1	1	
Brassicaceae	8	18	
Campanulaceae	1	1	
Caryophyllaceae	9	18	
Chenopodiaceae	1	1	
Elaeagnaceae	1	1	
Ericaceae	10	14	
Fabaceae	5	11	
Gentianaceae	4	4	
Haloragaceae	1	1	
Lentibulariaceae	2	4	
Linaceae	1	1	
Linnaeaceae	1	1	
Onagraceae	2	6	
Orobanchaceae	2	11	
Papaveraceae	1	1	
Parnassiaceae	1	2	
Plantaginaceae	2	3	
Plumbaginaceae	1	1	
Polygonaceae	5	5	
Primulaceae	2	4	
Ranunculaceae	5	11	
Rosaceae	5	12 (1,1)	
Salicaceae	2	13 (2,2)	
Saxifragaceae	3	10	
Total	93	181 (187)	
Grand Total	134	300 (311)	
Note:

In parentheses after each higher taxon name are the numbers of families in the higher taxon. In parentheses after number of species are numbers of infraspecific taxa and hybrids.

Table 2 Checklist of the vascular plant flora of the lower Coppermine River valley and vicinity.

The table records whether or not each taxon is newly recorded for Nunavut, mainland Nunavut, and/or the study area; if records are a range extension for a taxon; if a taxon has been previously recorded for the study area; if a taxon has been previously collected in the study area but was identified as a different taxon; if a taxon is recorded for Fockler Creek, Kugluk (Bloody Falls) Territorial Park, and/or Kugluktuk; and if a taxon is recorded for the Arctic and Subarctic portions of the study area. The number of unique collections from the study area examined for each taxon is also given.

Family	Taxon	Newly recorded for Nunavut	Newly recorded for mainland Nunavut	Newly recorded for study area	Range extension	Previously recorded	Fockler Creek	Kugluk/Bloody Falls Territorial Park	Kugluktuk	Arctic zone	Subarctic zone	Number of collections	Rank	
Gymnosperms	
Cupressaceae	Juniperus communis subsp. depressa				×	×	×	×		×	×	6	Species	
Pinaceae	Picea glauca					×	×	×		×	×	5	Species	
Lycophytes	
Lycopodiaceae	Huperzia arctica					×	×	×	×	×	×	5	Species	
Lycopodiaceae	Lycopodium annotinum subsp. alpestre			×			×				×	1	Species	
Monilophytes	
Cystopteridaceae	Cystopteris fragilis					×		×	×	×	×	5	Species	
Dryopteridaceae	Dryopteris fragrans					×	×	×	×	×	×	8	Species	
Equisetaceae	Equisetum arvense subsp. alpestre					×	×	×	×	×	×	6	Species	
Equisetaceae	Equisetum fluviatile			×	×			×	×	×		2	Species	
Equisetaceae	Equisetum palustre			×	×		×	×		×	×	5	Species	
Equisetaceae	Equisetum scirpoides			×	×		×	×		×	×	5	Species	
Equisetaceae	Equisetum variegatum subsp. variegatum					×	×	×	×*	×	×	5	Species	
Ophioglossaceae	Botrychium minganense			×						×		1	Species	
Ophioglossaceae	Botrychium neolunaria	×	×	×	×						×	1	Species	
Ophioglossaceae	Botrychium tunux	×	×	×	×					×		3	Species	
Pteridaceae	Cryptogramma stelleri		×	×	×			×		×		1	Species	
Woodsiaceae	Woodsia alpina					×			×	×		3	Species	
Woodsiaceae	Woodsia glabella					×	×	×	×	×	×	7	Species	
Woodsiaceae	Woodsia ilvensis					×			×	×		1	Species	
Monocots	
Amaryllidaceae	Allium schoenoprasum	×	×	×	×			×	×	×	×	3	Species	
Cyperaceae	Carex adelostoma			×	×		×			×	×	2	Species	
Cyperaceae	Carex aquatilis subsp. stans					×	×	×	×	×	×	6	Species	
Cyperaceae	Carex atrofusca					×	×	×	×	×	×	6	Species	
Cyperaceae	Carex bicolor			×			×	×	×	×	×	10	Species	
Cyperaceae	Carex bigelowii subsp. bigelowii					×		×	×	×		3	Species	
Cyperaceae	Carex bigelowii subsp. lugens					×	×	×	×	×	×	8	Subsp.	
Cyperaceae	Carex borealipolaris					×	×	×	×	×	×	6	Species	
Cyperaceae	Carex capillaris subsp. fuscidula					×	×	×	×	×	×	10	Species	
Cyperaceae	Carex capitata			×	×						×	2	Species	
Cyperaceae	Carex chordorrhiza					×	×		×	×	×	4	Species	
Cyperaceae	Carex concinna			×	×		×	×		×	×	6	Species	
Cyperaceae	Carex fuliginosa subsp. misandra					×	×	×	×	×	×	3	Species	
Cyperaceae	Carex glacialis					×	×	×	×	×	×	5	Species	
Cyperaceae	Carex glareosa subsp. glareosa			×						×		2	Species	
Cyperaceae	Carex gynocrates		×	×	×		×	×		×	×	2	Species	
Cyperaceae	Carex holostoma					×	×	×	×	×	×	5	Species	
Cyperaceae	Carex krausei			×			×	×	×	×	×	6	Species	
Cyperaceae	Carex lachenalii			×	×		×		×	×	×	3	Species	
Cyperaceae	Carex livida		×	×	×		×				×	1	Species	
Cyperaceae	Carex marina					×	×	×	×	×	×	8	Species	
Cyperaceae	Carex maritima					×		×	×	×		4	Species	
Cyperaceae	Carex membranacea					×	×	×	×	×	×	5	Species	
Cyperaceae	Carex microglochin			×			×	×	×	×	×	5	Species	
Cyperaceae	Carex myosuroides					×	×	×	×	×	×	5	Species	
Cyperaceae	Carex nardina			×			×	×	×	×	×	4	Species	
Cyperaceae	Carex norvegica			×	×		×	×		×	×	2	Species	
Cyperaceae	Carex petricosa subsp. petricosa			×			×	×		×	×	6	Species	
Cyperaceae	Carex podocarpa				×	×	×	×	×	×	×	12	Species	
Cyperaceae	Carex rariflora					×	×	×	×	×	×	4	Species	
Cyperaceae	Carex rupestris					×	×	×	×	×	×	7	Species	
Cyperaceae	Carex saxatilis					×	×	×	×	×	×	5	Species	
Cyperaceae	Carex scirpoidea subsp. scirpoidea					×	×	×	×	×	×	6	Species	
Cyperaceae	Carex simpliciuscula subsp. subholarctica					×	×	×	×*	×	×	5	Species	
Cyperaceae	Carex subspathacea			×						×		2	Species	
Cyperaceae	Carex supina subsp. spaniocarpa					×	×			×	×	6	Species	
Cyperaceae	Carex ursina			×						×		1	Species	
Cyperaceae	Carex vaginata					×	×	×	×	×	×	5	Species	
Cyperaceae	Carex williamsii			×						×		2	Species	
Cyperaceae	Eleocharis acicularis					×*				×	×	2	Species	
Cyperaceae	Eleocharis quinqueflora	×	×	×	×			×	×	×		2	Species	
Cyperaceae	Eriophorum angustifolium					×	×	×	×	×	×	7	Species	
Cyperaceae	Eriophorum brachyantherum					×	×		×	×	×	4	Species	
Cyperaceae	Eriophorum callitrix			×			×	×		×	×	3	Species	
Cyperaceae	Eriophorum scheuchzeri subsp. arcticum					×	×	×	×	×	×	12	Species	
Cyperaceae	Eriophorum triste			×			×	×		×	×	3	Species	
Cyperaceae	Eriophorum vaginatum subsp. vaginatum					×	×	×	×	×	×	4	Species	
Cyperaceae	Trichophorum cespitosum subsp. cespitosum					×	×	×	×	×	×	3	Species	
Juncaceae	Juncus alpinoarticulatus subsp. americanus		×	×	×			×		×		4	Species	
Juncaceae	Juncus arcticus subsp. alaskanus			×			×	×	×	×	×	6	Species	
Juncaceae	Juncus biglumis					×*		×	×*	×		1	Species	
Juncaceae	Juncus leucochlamys					×*	×	×	×*	×	×	3	Species	
Juncaceae	Juncus triglumis subsp. albescens					×*	×	×	×*	×	×	3	Species	
Juncaceae	Luzula confusa					×	×	×	×	×	×	7	Species	
Juncaceae	Luzula groenlandica					×*				×		1	Species	
Juncaceae	Luzula nivalis					×	×	×	×	×	×	5	Species	
Juncaginaceae	Triglochin maritima			×			×	×	×	×	×	3	Species	
Juncaginaceae	Triglochin palustris			×			×	×		×	×	5	Species	
Orchidaceae	Corallorhiza trifida					×	×	×	×	×	×	9	Species	
Orchidaceae	Platanthera obtusata subsp. obtusata					×	×	×	×	×	×	9	Species	
Poaceae	Agrostis mertensii			×	×						×	1	Species	
Poaceae	Alopecurus borealis					×		×	×	×		8	Species	
Poaceae	Anthoxanthum arcticum			×	×					×		1	Species	
Poaceae	Anthoxanthum hirtum			×	×				×	×		1	Species	
Poaceae	Anthoxanthum monticola subsp. alpinum					×	×	×	×	×	×	10	Species	
Poaceae	Arctagrostis latifolia subsp. arundinacea			×					×	×		2	Subsp.	
Poaceae	Arctagrostis latifolia subsp. latifolia					×	×	×		×	×	7	Species	
Poaceae	Arctophila fulva					×	×	×	×	×	×	8	Species	
Poaceae	Bromus pumpellianus					×	×	×	×	×	×	11	Species	
Poaceae	Calamagrostis canadensis subsp. langsdorffii			×				×	×	×	×	3	Species	
Poaceae	Calamagrostis lapponica			×			×	×	×	×	×	4	Species	
Poaceae	Calamagrostis purpurascens subsp. purpurascens					×	×	×	×	×	×	9	Species	
Poaceae	Calamagrostis stricta subsp. groenlandica			×			×	×		×	×	5	Species	
Poaceae	Calamagrostis stricta subsp. stricta			×	×			×		×		2	Subsp.	
Poaceae	Deschampsia brevifolia					×		×		×		2	Species	
Poaceae	Deschampsia cespitosa subsp. cespitosa					×	×	×	×*	×	×	9	Species	
Poaceae	Deschampsia sukatschewii subsp. borealis			×			×				×	1	Species	
Poaceae	Dupontia fisheri					×		×	×	×		7	Species	
Poaceae	Elymus alaskanus subsp. alaskanus			×			×	×	×	×	×	6	Species	
Poaceae	Elymus alaskanus subsp. hyperarcticus			×				×		×	×	4	Subsp.	
Poaceae	Elymus violaceus			×				×		×	×	2	Species	
Poaceae	Festuca altaica	×	×	×	×		×	×		×	×	4	Species	
Poaceae	Festuca baffinensis					×			×	×		2	Species	
Poaceae	Festuca brachyphylla subsp. brachyphylla					×	×	×	×	×	×	8	Species	
Poaceae	Festuca rubra subsp. arctica					×*	×	×	×	×	×	7	Subsp.	
Poaceae	Festuca rubra subsp. rubra			×				×	×	×		3	Species	
Poaceae	Festuca viviparoidea subsp. viviparoidea		×	×	×			×		×		1	Species	
Poaceae	Hordeum jubatum subsp. intermedium			×				×	×	×		4	Species	
Poaceae	Leymus mollis subsp. villosissimus					×			×	×		8	Species	
Poaceae	Phippsia algida			×				×		×		1	Species	
Poaceae	Poa alpina			×			×		×	×	×	6	Species	
Poaceae	Poa arctica subsp. arctica					×	×	×	×*	×	×	4	Species	
Poaceae	Poa glauca subsp. glauca					×*	×	×	×	×	×	13	Species	
Poaceae	Poa pratensis subsp. alpigena			×			×	×	×	×	×	10	Species	
Poaceae	Puccinellia arctica			×				×		×		3	Species	
Poaceae	Puccinellia nuttalliana			×			×	×	×	×	×	13	Species	
Poaceae	Puccinellia phryganodes subsp. neoarctica					×*			×*	×		1	Species	
Poaceae	Puccinellia vaginata			×					×	×	×	4	Species	
Poaceae	Trisetum spicatum					×	×	×	×	×	×	9	Species	
Potamogetonaceae	Potamogeton gramineus			×	×				×	×		1	Species	
Potamogetonaceae	Stuckenia filiformis			×				×		×		4	Species	
Potamogetonaceae	Stuckenia pectinata	×	×	×	×			×	×	×		4	Species	
Potamogetonaceae	Stuckenia vaginata			×				×		×	×	3	Species	
Tofieldiaceae	Tofieldia coccinea					×	×	×	×	×	×	8	Species	
Tofieldiaceae	Tofieldia pusilla					×	×	×	×	×	×	8	Species	
Typhaceae	Sparganium hyperboreum			×						×		1	Species	
Eudicots	
Asteraceae	Achillea millefolium subsp. borealis					×		×	×	×		8	Species	
Asteraceae	Antennaria friesiana—Antennaria alpina complex					×	×	×	×	×	×	11	Species	
Asteraceae	Antennaria monocephala subsp. angustata					×	×	×	×	×	×	7	Species	
Asteraceae	Arnica angustifolia subsp. angustifolia					×	×	×	×	×	×	5	Species	
Asteraceae	Arnica frigida					×		×	×	×	×	6	Species	
Asteraceae	Artemisia borealis subsp. borealis			×			×				×	2	Species	
Asteraceae	Artemisia hyperborea					×*		×	×*	×	×	2	Species	
Asteraceae	Artemisia tilesii					×	×	×	×	×	×	8	Species	
Asteraceae	Askellia pygmaea			×			×	×		×	×	5	Species	
Asteraceae	Erigeron eriocephalus					×	×		×	×		4	Species	
Asteraceae	Erigeron humilis					×	×	×	×	×	×	11	Species	
Asteraceae	Eurybia sibirica					×	×	×	×	×	×	14	Species	
Asteraceae	Hulteniella integrifolia					×	×	×	×	×	×	8	Species	
Asteraceae	Petasites frigidus subsp. sagittatus			×			×				×	1	Species	
Asteraceae	Saussurea angustifolia subsp. angustifolia					×	×	×	×	×	×	7	Species	
Asteraceae	Senecio lugens					×	×	×	×	×	×	8	Species	
Asteraceae	Symphyotrichum pygmaeum					×	×	×	×	×	×	8	Species	
Asteraceae	Taraxacum ceratophorum					×	×	×	×	×	×	17	Species	
Asteraceae	Taraxacum holmenianum			×	×				×	×		2	Species	
Asteraceae	Taraxacum phymatocarpum				×	×	×		×	×	×	4	Species	
Asteraceae	Tephroseris frigida					×	×	×	×	×	×	6	Species	
Asteraceae	Tephroseris palustris subsp. congesta					×			×	×		6	Species	
Asteraceae	Tripleurospermum maritimum subsp. phaeocephalum					×			×	×		2	Species	
Betulaceae	Alnus alnobetula			×	×						×	1	Species	
Betulaceae	Betula glandulosa					×	×	×	×	×	×	9	Species	
Betulaceae	Betula occidentalis			×	×			×	×	×	×	8	Species	
Boraginaceae	Mertensia maritima subsp. tenella					×				×		3	Species	
Brassicaceae	Braya glabella subsp. glabella					×	×				×	2	Species	
Brassicaceae	Braya humilis subsp. humilis			×			×	×		×	×	8	Species	
Brassicaceae	Cardamine bellidifolia			×			×				×	2	Species	
Brassicaceae	Cardamine digitata					×	×	×	×	×	×	6	Species	
Brassicaceae	Cardamine nymanii					×	×	×	×	×	×	6	Species	
Brassicaceae	Descurainia sophioides					×	×		×	×	×	13	Species	
Brassicaceae	Draba cinerea					×	×		×	×	×	10	Species	
Brassicaceae	Draba fladnizensis					×		×	×	×	×	6	Species	
Brassicaceae	Draba glabella					×	×	×	×	×	×	25	Species	
Brassicaceae	Draba lonchocarpa	×	×	×	×		×	×		×	×	6	Species	
Brassicaceae	Draba nivalis			×					×	×		2	Species	
Brassicaceae	Draba pilosa					×	×	×	×	×	×	7	Species	
Brassicaceae	Draba simmonsii		×	×	×		×				×	1	Species	
Brassicaceae	Erysimum coarctatum			×			×	×		×	×	10	Species	
Brassicaceae	Erysimum pallasii			×						×		1	Species	
Brassicaceae	Eutrema edwardsii			×			×				×	2	Species	
Brassicaceae	Physaria arctica			×						×	×	3	Species	
Brassicaceae	Transberingia bursifolia					×			×	×		2	Species	
Campanulaceae	Campanula uniflora			×			×	×		×	×	4	Species	
Caryophyllaceae	Arenaria humifusa					×	×		×	×	×	4	Species	
Caryophyllaceae	Cerastium alpinum subsp. alpinum					×	×	×	×	×	×	4	Species	
Caryophyllaceae	Cerastium beeringianum					×	×	×	×	×	×	10	Species	
Caryophyllaceae	Eremogone capillaris subsp. capillaris	×	×	×	×						×	1	Species	
Caryophyllaceae	Honckenya peploides subsp. diffusa					×			×	×		5	Species	
Caryophyllaceae	Minuartia biflora					×	×		×*	×	×	5	Species	
Caryophyllaceae	Sabulina elegans	×	×	×	×		×		×	×	×	4	Species	
Caryophyllaceae	Sabulina rossii			×			×				×	1	Species	
Caryophyllaceae	Sabulina rubella					×	×	×	×	×	×	15	Species	
Caryophyllaceae	Sabulina stricta			×			×				×	5	Species	
Caryophyllaceae	Sagina nodosa subsp. borealis			×	×			×	×	×		2	Species	
Caryophyllaceae	Silene acaulis					×	×	×	×	×	×	10	Species	
Caryophyllaceae	Silene involucrata subsp. tenella					×	×	×	×	×	×	5	Species	
Caryophyllaceae	Silene uralensis subsp. uralensis					×	×	×	×	×	×	8	Species	
Caryophyllaceae	Stellaria borealis subsp. borealis			×	×		×				×	1	Species	
Caryophyllaceae	Stellaria crassifolia			×	×		×	×		×		2	Species	
Caryophyllaceae	Stellaria humifusa					×*			×*	×		2	Species	
Caryophyllaceae	Stellaria longipes					×	×	×	×	×	×	15	Species	
Chenopodiaceae	Suaeda calceoliformis					×				×		2	Species	
Elaeagnaceae	Shepherdia canadensis			×	×		×	×		×	×	7	Species	
Ericaceae	Andromeda polifolia					×	×	×	×	×	×	9	Species	
Ericaceae	Arctostaphylos uva-ursi			×	×						×	1	Species	
Ericaceae	Arctous alpina					×			×	×		4	Species	
Ericaceae	Arctous rubra					×	×	×	×	×	×	4	Species	
Ericaceae	Cassiope tetragona subsp. tetragona					×	×	×	×	×	×	9	Species	
Ericaceae	Empetrum nigrum					×	×	×	×	×	×	6	Species	
Ericaceae	Kalmia procumbens					×*			×*	×*	×	1	Species	
Ericaceae	Orthilia secunda subsp. obtusata					×	×	×	×	×	×	8	Species	
Ericaceae	Pyrola grandiflora subsp. grandiflora					×	×	×	×	×	×	8	Species	
Ericaceae	Rhododendron groenlandicum			×	×		×	×		×	×	5	Species	
Ericaceae	Rhododendron lapponicum					×	×	×	×	×	×	5	Species	
Ericaceae	Rhododendron tomentosum subsp. decumbens					×	×	×	×	×	×	8	Species	
Ericaceae	Vaccinium uliginosum					×	×	×	×	×	×	8	Species	
Ericaceae	Vaccinium vitis-idaea subsp. minus					×	×	×	×	×	×	6	Species	
Fabaceae	Astragalus alpinus					×	×	×	×	×	×	12	Species	
Fabaceae	Astragalus richardsonii					×	×	×	×	×	×	11	Species	
Fabaceae	Hedysarum americanum					×	×	×	×	×	×	9	Species	
Fabaceae	Hedysarum boreale subsp. mackenziei					×	×	×		×	×	6	Species	
Fabaceae	Lathyrus japonicus					×	×			×		5	Species	
Fabaceae	Lupinus arcticus					×	×	×	×	×	×	9	Species	
Fabaceae	Oxytropis arctica					×	×		×	×	×	4	Species	
Fabaceae	Oxytropis arctobia					×		×		×		4	Species	
Fabaceae	Oxytropis deflexa subsp. foliolosa					×	×	×	×	×	×	3	Species	
Fabaceae	Oxytropis maydelliana					×	×	×	×	×	×	8	Species	
Fabaceae	Oxytropis varians					×	×	×	×*	×	×	4	Species	
Gentianaceae	Comastoma tenellum					×*			×	×		1	Species	
Gentianaceae	Gentianella propinqua subsp. propinqua					×	×	×	×	×	×	10	Species	
Gentianaceae	Gentianopsis detonsa subsp. detonsa					×			×	×		2	Species	
Gentianaceae	Lomatogonium rotatum subsp. rotatum					×				×		1	Species	
Haloragaceae	Myriophyllum sibiricum			×				×		×		1	Species	
Lentibulariaceae	Pinguicula villosa			×	×		×			×	×	2	Species	
Lentibulariaceae	Pinguicula vulgaris					×	×	×	×	×	×	6	Species	
Lentibulariaceae	Utricularia intermedia			×	×		×				×	1	Species	
Lentibulariaceae	Utricularia vulgaris			×	×			×		×		1	Species	
Linaceae	Linum lewisii subsp. lewisii			×							×	1	Species	
Linnaeaceae	Linnaea borealis subsp. americana			×	×			×		×	×	5	Species	
Onagraceae	Chamerion angustifolium subsp. angustifolium					×	×	×	×	×	×	6	Species	
Onagraceae	Chamerion latifolium					×	×	×	×	×	×	9	Species	
Onagraceae	Epilobium anagallidifolium	×	×	×	×				×	×		1	Species	
Onagraceae	Epilobium arcticum			×	×			×		×		1	Species	
Onagraceae	Epilobium davuricum			×	×			×		×		1	Species	
Onagraceae	Epilobium palustre					×		×	×	×		5	Species	
Orobanchaceae	Castilleja caudata			×			×	×	×	×	×	11	Species	
Orobanchaceae	Castilleja elegans					×			×	×		7	Species	
Orobanchaceae	Castilleja raupii			×	×			×	×	×	×	3	Species	
Orobanchaceae	Pedicularis albolabiata					×	×	×	×	×	×	11	Species	
Orobanchaceae	Pedicularis arctoeuropaea			×					×	×		2	Species	
Orobanchaceae	Pedicularis capitata					×	×	×	×	×	×	6	Species	
Orobanchaceae	Pedicularis flammea			×			×				×	1	Species	
Orobanchaceae	Pedicularis labradorica					×		×	×*	×	×	4	Species	
Orobanchaceae	Pedicularis lanata					×	×	×	×	×	×	7	Species	
Orobanchaceae	Pedicularis langsdorffii subsp. arctica					×	×	×	×	×	×	7	Species	
Orobanchaceae	Pedicularis lapponica					×	×	×	×	×	×	8	Species	
Papaveraceae	Papaver hultenii					×	×	×	×	×	×	11	Species	
Parnassiaceae	Parnassia kotzebuei					×	×	×	×	×	×	13	Species	
Parnassiaceae	Parnassia palustris subsp. neogaea					×		×	×	×		7	Species	
Plantaginaceae	Hippuris lanceolata			×						×		1	Species	
Plantaginaceae	Hippuris vulgaris			×				×	×	×		3	Species	
Plantaginaceae	Plantago canescens subsp. richardsonii					×	×	×	×	×	×	6	Species	
Plumbaginaceae	Armeria maritima subsp. sibirica					×	×	×	×	×	×	13	Species	
Polygonaceae	Bistorta vivipara					×	×	×	×	×	×	7	Species	
Polygonaceae	Koenigia islandica					×			×	×		3	Species	
Polygonaceae	Oxyria digyna					×	×	×	×	×	×	8	Species	
Polygonaceae	Polygonum aviculare	×	×	×	×				×	×		1	Species	
Polygonaceae	Rumex arcticus					×	×	×	×	×	×	8	Species	
Primulaceae	Androsace chamaejasme subsp. andersonii					×			×	×		2	Species	
Primulaceae	Androsace septentrionalis					×	×		×	×	×	7	Species	
Primulaceae	Primula egaliksensis					×	×	×	×	×	×	8	Species	
Primulaceae	Primula stricta					×*				×		1	Species	
Ranunculaceae	Anemone parviflora					×	×	×	×	×	×	3	Species	
Ranunculaceae	Anemone richardsonii					×		×	×	×	×	6	Species	
Ranunculaceae	Caltha palustris subsp. radicans					×		×	×	×		7	Species	
Ranunculaceae	Coptidium pallasii					×			×	×		1	Species	
Ranunculaceae	Halerpestes cymbalaria					×				×		3	Species	
Ranunculaceae	Ranunculus arcticus					×		×	×	×	×	5	Species	
Ranunculaceae	Ranunculus confervoides			×	×			×		×		1	Species	
Ranunculaceae	Ranunculus gmelinii subsp. gmelinii					×			×	×		4	Species	
Ranunculaceae	Ranunculus hyperboreus subsp. hyperboreus					×*			×	×		2	Species	
Ranunculaceae	Ranunculus nivalis					×*	×		×*	×*	×	2	Species	
Ranunculaceae	Ranunculus pygmaeus					×	×		×	×	×	3	Species	
Rosaceae	Comarum palustre			×					×	×		3	Species	
Rosaceae	Dasiphora fruticosa					×	×	×	×	×	×	10	Species	
Rosaceae	Dryas integrifolia subsp. integrifolia					×	×	×	×	×	×	7	Species	
Rosaceae	Potentilla anserina subsp. groenlandica			×						×		1	Species	
Rosaceae	Potentilla arenosa subsp. arenosa			×			×	×	×	×	×	7	Species	
Rosaceae	Potentilla arenosa subsp. chamissonis					×			×	×		1	Subsp.	
Rosaceae	Potentilla arenosa subsp. chamissonis x Potentilla nivea			×			×		×	×	×	4	Hybrid	
Rosaceae	Potentilla biflora			×			×				×	3	Species	
Rosaceae	Potentilla hyparctica subsp. hyparctica			×			×				×	2	Species	
Rosaceae	Potentilla nivea					×	×	×	×	×	×	8	Species	
Rosaceae	Potentilla pulchella			×						×		1	Species	
Rosaceae	Potentilla tikhomirovii			×			×	×	×	×	×	8	Species	
Rosaceae	Rubus arcticus subsp. acaulis			×	×		×	×		×	×	4	Species	
Rosaceae	Rubus chamaemorus					×	×	×	×	×	×	6	Species	
Salicaceae	Populus balsamifera					×	×			×	×	2	Species	
Salicaceae	Salix alaxensis var. alaxensis					×	×	×	×	×	×	6	Species	
Salicaceae	Salix arbusculoides			×	×		×				×	1	Species	
Salicaceae	Salix arctica					×	×	×	×	×	×	18	Species	
Salicaceae	Salix arctica x Salix arctophila			×					×	×		2	Hybrid	
Salicaceae	Salix arctophila					×			×	×		5	Species	
Salicaceae	Salix glauca var. cordifolia					×	×	×	×	×	×	28	Subsp.	
Salicaceae	Salix glauca var. glauca			×				×		×		2	Species	
Salicaceae	Salix niphoclada					×	×	×	×	×	×	20	Species	
Salicaceae	Salix niphoclada x Salix glauca			×				×		×		2	Hybrid	
Salicaceae	Salix ovalifolia var. arctolitoralis	×	×	×	×				×	×		2	Species	
Salicaceae	Salix ovalifolia var. ovalifolia	×	×	×	×					×		1	Subsp.	
Salicaceae	Salix planifolia			×	×		×	×		×	×	4	Species	
Salicaceae	Salix pseudomyrsinites		×	×	×					×		1	Species	
Salicaceae	Salix pulchra					×		×	×	×	×	7	Species	
Salicaceae	Salix reticulata					×	×	×	×	×	×	10	Species	
Salicaceae	Salix richardsonii					×	×	×	×	×	×	15	Species	
Saxifragaceae	Chrysosplenium rosendahlii			×			×	×	×	×	×	8	Species	
Saxifragaceae	Micranthes foliolosa					×	×		×	×	×	5	Species	
Saxifragaceae	Micranthes nivalis					×	×	×	×	×	×	8	Species	
Saxifragaceae	Micranthes porsildiana			×					×	×		2	Species	
Saxifragaceae	Saxifraga aizoides					×	×	×	×	×	×	5	Species	
Saxifragaceae	Saxifraga cernua					×	×	×	×	×	×	14	Species	
Saxifragaceae	Saxifraga hirculus					×	×	×	×	×	×	9	Species	
Saxifragaceae	Saxifraga hyperborea					×	×		×	×	×	2	Species	
Saxifragaceae	Saxifraga oppositifolia subsp. oppositifolia					×	×	×	×	×	×	6	Species	
Saxifragaceae	Saxifraga tricuspidata					×	×	×	×	×	×	6	Species	
	Total	13	20	121	56	190	199	207	215	287	223	1,653		
Note:

An asterisk indicates that we were unable to find a voucher specimen corresponding to a literature record.

A total of 22 families (49%) are represented by one genus, 18 families (40%) by 2–5, and five families (11%) by 8–17 (Table 1). The three families with the largest number of genera are Ericaceae (10 genera), Asteraceae (15 genera), and Poaceae (17 genera). A total of 16 families include a single species, seventeen 2–8, seven 11–14, and three more than 20 species (Table 1). The latter are Asteraceae (23 species), Poaceae (35 species), and Cyperaceae (46 species). Of the three hybrids recorded, two are in Salix L. (Salicaceae) and one in Potentilla L. (Rosaceae). Of the eight subspecies or varieties (i.e., in species with more than one infraspecific taxon recorded in the study area), one each is in Arctagrostis Griseb., Calamagrostis Adans., Elymus L., Festuca L. (Poaceae), Carex L. (Cyperaceae) and Potentilla, and two are in Salix (Salicaceae).

Based on our study of all previous and new collections from the study area, we found that 190 taxa (61% of the taxa now known in the region) were previously recorded, and 121 taxa (39%) are newly recorded (Table 2). The majority of the taxa new to the study area are documented by collections we made in 2014; however, 11 of these were first collected prior to 2014 and re-determined during this study as taxa not previously recorded for the area. These include Calamagrostis stricta subsp. groenlandica (Schrank) Á. Löve, Castilleja caudata (Pennell) Rebrist., Chrysosplenium rosendahlii Packer, Erysimum coarctatum Fernald, Pedicularis arctoeuropaea (Hultén) Molau & D. F. Murray, Potentilla arenosa (Turcz.) Juz. subsp. arenosa, Potentilla tikhomirovii Juz., Sabulina elegans (Cham. & Schltdl.) Dillenb. & Kadereit, Salix arctica Pall. × Salix arctophila Cockerell ex A. Heller and Taraxacum holmenianum Sahlin. We also made new collections of these taxa. A total of 14 taxa (13 species and one additional variety) are newly recorded for Nunavut (Allium schoenoprasum L., Botrychium neolunaria, Botrychium tunux Stensvold & Farrar, Carex capitata L., Draba lonchocarpa Rydb., Eleocharis quinqueflora (Hartmann) O. Schwarz, Epilobium cf. anagallidifolium Lam., Eremogone capillaris (Poir.) Fenzl subsp. capillaris, Festuca altaica Trin., Polygonum aviculare L., Sabulina elegans, Salix ovalifolia var. arctolitoralis (Hultén) Argus, Salix ovalifolia Trautv. var. ovalifolia and Stuckenia pectinata (L.) Börner) (Table 2). Seven species are newly recorded for mainland Nunavut (in addition to the 13 taxa new to Nunavut): Carex gynocrates Wormsk. ex Drejer, Carex livida (Wahlenb.) Willd., Cryptogramma stelleri (S. G. Gmel.) Prantl., Draba simmonsii Elven & Al-Shehbaz, Festuca viviparoidea Krajina ex Pavlick subsp. viviparoidea, Juncus alpinoarticulatus subsp. americanus (Farwell) Hämet-Ahti and Salix pseudomyrsinites Andersson (Table 2). Fifty-six range extensions are recorded (including the species new to Nunavut and mainland Nunavut): Agrostis mertensii Trin., Alnus alnobetula (Ehrh.) K. Koch, Anthoxanthum arcticum Veldkamp, Anthoxanthum hirtum (Schrank) Y. Schouten & Veldkamp, Arctostaphylos uva-ursi (L.) Spreng., Betula occidentalis Hook., Calamagrostis stricta (Timm) Koeler subsp. stricta, Carex adelostoma V. I. Krecz., Carex capitata, Carex concinna, Carex lachenalii Schkuhr, Carex norvegica Retz., Carex podocarpa, Castilleja raupii Pennell, Epilobium arcticum Sam., Epilobium davuricum Fisch. ex Horn., Equisetum fluviatile L., Equisetum palustre L., Equisetum scirpoides Michx., Juniperus communis subsp. depressa (Pursh) Franco, Linnaea borealis subsp. americana (J. Forbes) Hultén, Potamogeton gramineus L., Ranunculus confervoides (Fr.) Fr., Rhododendron groenlandicum (Oeder) Kron & Judd, Rubus arcticus subsp. acaulis (Michx.) Focke, Sagina nodosa subsp. borealis G. E. Crow, Salix arbusculoides Andersson, Salix planifolia Pursh, Shepherdia canadensis (L.) Nutt., Stellaria borealis Bigelow subsp. borealis, Stellaria crassifolia Ehrh., Taraxacum holmenianum, Taraxacum phymatocarpum J. Vahl, Utricularia intermedia Hayne, and Utricularia vulgaris L. (Table 2). A total of 287 taxa are recorded for the Arctic portion of the study area and 223 for the Subarctic portion. A total of 199 taxa are recorded from both ecozones, 88 taxa are recorded only in the Arctic portion of the study area and 24 taxa are recorded only in the Subarctic portion of the study area. We recorded 207 taxa in Kugluk (Bloody Falls) Territorial Park (201 species, five subspecies, and one hybrid), 215 taxa (208 species, five subspecies, and two hybrids) at Kugluktuk, and 199 taxa (195 species, three subspecies, and one hybrid) in the Fockler Creek area. Of the 311 taxa recorded in the entire study area, 29 were recorded outside of these three sites.

Discussion

We recorded 300 species and 311 taxa along the lower Coppermine River valley and vicinity in western Nunavut, in an area extending from the forest-tundra to the Arctic coast. Of these, 288 species are known from the Arctic portion of the study area. This level of species diversity falls within the range expected for local floras for bioclimatic subzone E (200–500) in the global Arctic, of which the Arctic portion of the study area is part.

Knowledge of floristic diversity in an area depends on the level of search effort invested in combination with the skill levels of the individuals studying the flora. In general, known floristic diversity in large areas increases as botanists explore them through time. Despite being species-poor compared to other major global ecozones, the Arctic is no exception to this pattern, as demonstrated for Arctic areas that have been studied repeatedly and comprehensively. An example is Tuktut Nogait National Park and vicinity on mainland Northwest Territories and adjacent Nunavut, where we increased the known flora by some 16% compared to an earlier inventory (Saarela et al., 2013a). The current study also demonstrates this. We found that 190 species had been recorded in the study area, based on our examination of all materials previously collected there (219 unique collections). Our one month of intensive botanising in 2014 increased the known flora by 36% to 300 taxa, with 121 taxa newly recorded for the area, including 14 taxa new to Nunavut and seven new to mainland Nunavut.

Some of the species newly recorded for the study area are common and widespread Arctic or low Arctic species whose presence is not unexpected. Examples of these include Eriophorum callitrix Cham., Eriophorum triste (Th. Fr.) Hadač & Á. Löve, Equisetum scirpoides, Elymus alaskanus (Scribn. & Merr.) Á. Löve, Carex krausei Boeckeler, Carex nardina Fr., Juncus arcticus subsp. alaskanus Hultén and Poa alpina L. Other taxa newly recorded are known from only one or a few collections elsewhere in Nunavut, such as Betula occidentalis and Carex petricosa subsp. petricosa. Of the species newly recorded for Nunavut, the majority have amphi-Beringian distributions, at least in the Arctic portion of their ranges, known previously from sites in adjacent Northwest Territories (Draba lonchocarpa, Eremogone capillaris subsp. capillaris, Festuca altaica, Sabulina elegans, Salix ovalifolia var. arctolitoralis, Salix ovalifolia var. ovalifolia). Records of these in the study area represent new eastern or northeastern limits for the taxa. A few taxa new to Nunavut have more widespread distributions in North America (Allium schoenoprasum, Carex capitata, Eleocharis quinqueflora, Polygonum aviculare, Stuckenia pectinata), which now include western Nunavut. The Botrychium Sw. taxa new to the study area include a combination of recently described but poorly known taxa (Botrychium tunux) and a widespread taxon whose circumscription is somewhat unclear (Botrychium neolunaria).

The seven species newly recorded for mainland Nunavut were previously known in the territory from one or more eastern Arctic islands (Carex gynocrates, Cryptogramma stelleri, Draba simmonsii), a northern and a western Arctic island (Festuca viviparoidea subsp. viviparoidea) or at least one island in James Bay (Carex livida, Juncus alpinoarticulatus subsp. americanus, Salix pseudomyrsinites). The 54 range extensions are represented by records from areas beyond published ranges, including the 20 species new to Nunavut and mainland Nunavut. These include range extensions to the north (e.g., Carex podocarpa, Juniperus communis subsp. depressa, Linnaea borealis subsp. americana, Rhododendron groenlandicum, Shepherdia canadensis, Stellaria crassifolia), northeast (e.g., Agrostis mertensii, Carex concinna, Carex lachenalii, Castilleja raupii, Cryptogramma stelleri, Equisetum fluviatile, Rubus arcticus subsp. acaulis, Utricularia vulgaris), east (e.g., Festuca viviparoidea subsp. viviparoidea), and south (e.g., Anthoxanthum arcticum, Taraxacum holmenianum).

Our discovery of such a large number of species new to the area can be attributed to two main factors. First, our team of three spent more time (one month) in the study area and explored many more sites and habitats, from the Subarctic to the Arctic coast, compared to previous researchers, who mostly collected alone. With the exception of Findlay, who made a large plant collection, most previous collectors gathered a small number of collections in one or a few days, mostly from Kugluktuk, where the flora is not fully representative of the entire study area. Second, the sole purpose of our research trip was to comprehensively document the vascular plant flora of the area. By contrast, most previous collectors were not trained botanists and gathered plant collections incidental to other activities.

Only seven of the 190 species previously collected in the study area document species we did not find in 2014: Coptidium pallasii (Schltdl.) Tzvelev (collected in Kugluktuk in 1951), Festuca baffinensis Polunin (collected at Rae River, but see the taxon treatment for problematic collection details, and in Kugluktuk in 2013), Lomatogonium rotatum (L.) Fr. subsp. rotatum (mouth of the Rae River in 1955), Potentilla arenosa subsp. chamissonis (Hultén) Elven & D. F. Murray (Kugluktuk in 1940), Salix arctophila (Kugluktuk by different collectors in 1949, 1951, and 1962), Transberingia bursifolia (DC.) Al-Shehbaz & O’Kane (Kugluktuk in 1951) and Woodsia ilvensis (L.) R. Br. (Kugluktuk in 1958). Except Festuca baffinensis, none of these taxa has been collected in the area in over 50 years, and it is possible they are no longer present. Kugluktuk in the 1950s was much smaller than it is today, and these species may have occurred in areas that have since been altered through development. Alternatively, they may simply have been overlooked during our survey. Focused searches should be undertaken to determine their current status, particularly around Kugluktuk where most were collected before.

Introduced species

The vascular plant flora of the lower Coppermine River valley and vicinity is comprised almost entirely of native species. The only non-native taxon recorded in the study area is Festuca rubra L. subsp. rubra. It was likely planted in Kugluktuk, where we collected it in a residential yard and on the ball diamond, and may have been introduced unintentionally from the community to the fishing area at Bloody Falls, where we found it growing on the rocky ledges immediately above the falls. Two species native in northern North America may be introduced in the study area, but it is impossible to determine their statuses with confidence. Hordeum jubatum L., newly reported for the study area, is native to western North America and often grows like a weed; it may have established itself in the study area in recent times. It is known from other Arctic sites on mainland Nunavut and on Baffin Island (see species account). Another weedy species, Polygonum aviculare, newly recorded for the area, often grows like a weed, but may be native (see species account). Future surveyors should be on the lookout for introduced species, particularly in Kugluktuk, where disturbed habitats are abundant and introduced species may be most likely to get established.

Vascular plant diversity and mean July temperature

Levels of vascular plant species diversity in an area depend on numerous interacting factors, including climate, geology, topography, local habitat variation, site exposure, photoperiod, nutrient availability, moisture variability, soil variation, length of growing season, and the postglacial history of the flora. Rannie (1986) demonstrated a strong relationship (r = 0.97) between floristic diversity and mean July temperature at numerous Canadian Arctic localities with complete vascular plant inventories, and found a diversity gradient of 24–26 species per degree Celsius. It is possible to test this relationship with our empirical floristic richness data for Kugluktuk, where long term climate data are available. Kugluktuk has a 30-year (1985–2014) mean July temperature of 11.0 °C, as calculated from data from Kugluktuk A climate station (67°49′N, 115°08′38″) (Environment Canada, 2015).

Rannie’s (1986) two equations (where N = number of species and T = mean July temperature), one based strictly on temperature station data (N = 24.2T − 29.1) and the other based on combined temperature station and other temperature data (N = 25.8T − 41.8), predict 242 and 237 species for Kugluktuk, respectively. These estimates are 16.9 and 12.6% greater than the 207 species (not including hybrids or infraspecific taxa) we documented there, and both are outside the standard error of 12 species for the equations (Rannie, 1986). Unfortunately, Rannie (1986) did not define what constitutes a “locality” in her study and it is therefore unclear how to define a “locality” geographically when applying the formulas. This is important, as the number of species in an area generally increases with the size of an area. We defined the Kugluktuk area fairly narrowly here to cover some 3 km2, but if we broaden the limits to include the nearby areas (within 20 km) we explored west and northeast of the hamlet, species richness for that locality increases considerably. We recorded some 23 additional species from nearby Heart Lake and/or other sites west of Kugluktuk (e.g., Anthoxanthum arcticum, Carex williamsii Britton, Comarum palustre L., Eleocharis acicularis (L.) Roem. & Schult., Erysimum pallasii (Pursh) Fernald, Lathyrus japonicus Willd., Luzula groenlandica Böcher), the confluence of the Rae and Richardson rivers at Coronation Gulf (Suaeda calceoliformis (Hook.) Moq., Carex glareosa Wahlenb. subsp. glareosa, Carex subspathacea Wormsk, Carex ursina Dewey, Festuca baffinensis, Halerpestes cymbalaria (Pursh) Greene, Lomatogonium rotatum subsp. rotatum, Puccinellia vaginata (Lange) Fernald & Welsh, Potentilla anserina subsp. groenlandica Tratt., Potentilla pulchella R. Br., Triglochin palustris L.) and an island in the mouth of the Coppermine River (Hippuris lanceolata Retz., Mertensia maritima subsp. tenella (Th. Fr.) Elven & Skarpaas, Primula stricta Wormsk., Stuckenia filiformis (Pers.) Börner, Stuckenia vaginata (Turcz.) Holub.). Including these, species diversity for the Kugluktuk area increases to 230, a count only 5 and 3% lower than Rannie’s (1986) formulas estimate, respectively, for the area based on its mean July temperature, and well within the margin of error (12 species).

Mean July temperature data are not available for the other two sites we inventoried completely, Kugluk (Bloody Falls) Territorial Park in the Arctic ecozone and Fockler Creek in the Subarctic ecozone, but we assume the mean July temperatures would be higher at these sites because they are further inland (14 and 40 km, respectively) and away from the cooling influence of the coast. Under that assumption, our observed species diversity at Fockler Creek (195 species) and Kugluk (Bloody Falls) Territorial Park (201 species) would be considerably lower than Rannie’s (1986) formulas estimate. For example, with the equation based on temperature station data an 11.5 °C mean July temperature would predict 249 species and a 12 °C mean July temperature would predict 261 species. If we expanded the limits of these areas and/or spent more time exploring these areas, greater species diversity might be recorded in accordance with the formula predictions; or the relationship may not apply in areas like these with higher mean July temperatures.

The formula based on temperature station data was developed with data from 14 sites in the Canadian Arctic Archipelago and five in northern Quebec and Labrador, with the highest mean July temperature in the analysis being 9.3 °C, at Deception Bay, Quebec. The second formula, based on temperature station and other temperature data, was developed with data from more sites in the Canadian Arctic Archipelago and northern Quebec and Labrador. However, in that analysis, only a single site on mainland Nunavut/Northwest Territories site (Bathurst Inlet) was included, and it was the only one with a mean temperature of 10 °C or greater. Since both formulas overestimate floristic diversity for Kugluktuk and hypothetically for the other areas we surveyed in detail (with the degree of that overestimation varying depending on how the limits of the areas are defined), it is possible the relationship between mean July temperature and floristic diversity may not be as strong or accurate in areas, such as Kugluktuk, with higher average temperatures than those used to develop the formulas. Floristic data from additional sites in bioclimate subzone D are needed to test this; unfortunately such data do not exist. Elsewhere in the Arctic, the formula considerably overestimated diversity at Burwash Bay on Baffin Island (Jacobs et al., 1997), and species diversity and mean July temperature were not strongly correlated in a study of vascular plant species richness along the Hood River in Nunavut, attributed to considerable variation in other factors (Gould & Walker, 1997).

Comparisons of floristic diversity among ecozones and within the study area

The distribution maps in Porsild & Cody (1980) show numerous boreal species reaching or extending slightly beyond the treeline area across Northwest Territories and Nunavut, and numerous species distributed widely across the boreal forest and low Arctic areas. We thus expected species diversity to be higher in the Subarctic portion of the study area compared to the Arctic portion. However, more species were documented in the Arctic (287 species) than the Subarctic portion (223 species). This may be a function of the amount of search effort expended, as we spent more time at a greater number of sites within the Arctic ecozone compared to the Subarctic ecozone (Fig. 3). Nevertheless, there are considerable floristic differences between these ecozones considering presence and absence of species: 199 taxa were recorded in both, 88 were recorded only in the Arctic ecozone, and 24 were recorded only in the Subarctic ecozone. We found similar numbers of taxa in Kugluk (Bloody Falls) Territorial Park (207), Kugluktuk (215) and the Fockler Creek area (199), and each had a subset of unique species (Table 2).

Species found only in the Subarctic portion of the study area include boreal or primarily boreal species (Alnus alnobetula, Arctostaphylos uva-ursi, Botrychium neolunaria, Braya glabella subsp. glabella, Carex adelostoma, Carex capitata, Carex livida, Deschampsia sukatschewii subsp. borealis (Trautv.) Tzvelev, Eremogone capillaris subsp. capillaris, Petasites frigidus subsp. sagittatus, Potentilla biflora Willd. ex Schltdl., Sabulina stricta (Sw.) Rchb., Salix arbusculoides, Stellaria borealis subsp. borealis, Utricularia intermedia), species that span the boreal and low Arctic regions of North America (Agrostis mertensii, Artemisia borealis Pallas subsp. borealis, Linum lewisii Pursh subsp. lewisii, Lycopodium annotinum subsp. alpestre (Hartm.) Á. Löve & D. Löve) and primarily Arctic taxa (Cardamine bellidifolia L., Draba simmonsii, Eutrema edwardsii R. Br., Pedicularis flammea L., Potentilla hyparctica Malte subsp. hyparctica, Sabulina rossii (R. Br.) Dillenb. & Kadereit). All but three of the boreal or primarily boreal species (Carex adelostoma, Carex livida, Utricularia intermedia) have been recorded elsewhere in the Canadian Arctic ecozone, and all the other taxa recorded only from the Subarctic portion of the study area have been recorded elsewhere in Arctic areas. It is likely that many or possibly all of these taxa extend further north than currently documented in the study area.

We expected to find numerous, rare boreal species at or near the northern limits of their ranges at the Subarctic sites south of Fockler Creek, especially those (Kendall River, Bigtree River, Melville Creek) where white spruce forests were much more extensive than the forest stands at Fockler Creek, but we did not find this to be the case. Although we did not exhaustively survey the sites south of Fockler Creek and spent relatively short periods of time at each, we did search each site for unique or rare species. We found only six species at these southern sites that we did not also find at Fockler Creek or further north: Agrostis mertensii and Alnus alnobetula at the Bigtree River site, Eremogone capillaris subsp. capillaris at the Kendall River site, Carex capitata at the Melville Creek and Kendall River sites, Arctostaphylos uva-ursi at the Melville Creek site, and Botrychium neolunaria at the Coppermine Mountains site (an upland tundra habitat). Three of these (Arctostaphylos uva-ursi, Botrychium neolunaria, Eremogone capillaris subsp. capillaris) have not been recorded within the Arctic ecozone in Nunavut. The species growing in the understories of the white spruce forests and open habitats at Subarctic sites were generally the same common low Arctic species present in the tundra communities in the Arctic ecozone portion of the study area (J. M. Saarela, 2014, personal observation). This pattern has been observed elsewhere at or near treeline, such as on the Seward Peninsula of Alaska and in the forest-tundra around Nueltin Lake in southern Nunavut (Porsild, 1950b). Nevertheless, further exploration of the Subarctic white spruce forest and forest-tundra along the Coppermine River will undoubtedly result in discovery of additional locally rare species at or near their northern limits in Nunavut.

We documented 207 vascular plant taxa in Kugluk (Bloody Falls) Territorial Park (Table 2). We also collected nine additional species in the Bloody Falls area outside the park boundary. Three of these were found just beyond the park boundary on the west side of the river (Anthoxanthum arcticum, Eriophorum brachyantherum Trautv. & C. A. Mey., Leymus mollis subsp. villosissimus (Scribn.) Á. Löve & D. Löve) and the remainder were on the east side of the river (Botrychium tunux, Botrychium minganense Vict., Draba cinerea Adams, Halerpestes cymbalaria, Oxytropis varians (Rydb.) K. Schum., Physaria arctica (Wormsk. ex Hornem.) O’Kane & Al-Shehbaz, Woodsia alpina (Bolton) S. F. Gray). As such, the park and immediate surrounding area are home to a considerable amount of plant biodiversity along the lower Coppermine River, including some rare species known only from the Bloody Falls area within the study area. These include Anthoxanthum arcticum, Calamagrostis stricta subsp. stricta, Cryptogramma stelleri, Deschampsia brevifolia R. Br., Elymus violaceus (Hornem.) Feilberg, Epilobium arcticum, Festuca viviparoidea subsp. viviparoidea, Myriophyllum sibiricum Kom., Phippsia algida (Sol.) R. Br., Puccinellia arctica (Hook.) Fernald & Weath., Ranunculus confervoides (Fr.) Fr., Salix glauca L. var. glauca, and Stellaria crassifolia. Two of these records are notable as southern range extensions (Anthoxanthum arcticum and Epilobium arcticum).

Some 26 species were not collected at sites further north than Kugluk (Bloody Falls) Territorial Park (we collected all of these at more than one site in the study area). These include Artemisia hyperborea Rydb., Askellia pygmaea, Botrychium minganense, Botrychium tunux, Braya humilis (C. A. Mey.) B. L. Rob. subsp. humilis, Calamagrostis stricta subsp. groenlandica, Campanula uniflora L., Carex concinna, Carex norvegica, Carex petricosa subsp. petricosa, Castilleja raupii, Elymus alaskanus subsp. alaskanus, Elymus alaskanus subsp. hyperarcticus (Polunin) Á. Löve & D. Löve, Equisetum palustre, Equisetum scirpoides, Eriophorum callitrix, Equisetum triste, Festuca altaica, Juniperus communis subsp. depressa, Linnaea borealis subsp. americana, Physaria arctica, Picea glauca, Shepherdia canadensis, Stuckenia vaginata, Rhododendron groenlandicum, Rubus arcticus subsp. acaulis and Utricularia vulgaris. A subset of these taxa are at their known northern limits in Nunavut in Kugluk (Bloody Falls) Territorial Park and vicinity: Artemisia hyperborea, Botrychium tunux, Carex concinna, Carex norvegica, Carex petricosa subsp. petricosa, Castilleja raupii, Cryptogramma stelleri, Festuca altaica, Juniperus communis subsp. depressa, Linnaea borealis subsp. americana, Shepherdia canadensis, Rhododendron groenlandicum, Rubus arcticus subsp. acaulis and Utricularia vulgaris. We did not explore the flora along the Coppermine River between the northern boundary of Kugluk (Bloody Falls) Territorial Park and just south of Kugluktuk (approximately 67°48′54″N, 115°6′9.1″W) and it is likely that at least some of these taxa occur in this area north of the park. Further field work is required to determine their true limits in the area, and to characterise the environmental conditions limiting their local distributions. The conspicuous shrub species Juniperus communis and Shepherdia canadensis, which are both fairly common along the west banks of the Coppermine River in the park north of Bloody Falls, would be particularly easy to survey along the banks of the Coppermine River between the park and Kugluktuk. Even if some or all of these taxa are eventually found to occur north of the park, the current data identify Kugluk (Bloody Falls) Territorial Park as an important conservation area for many regionally rare vascular plant species.

In the northern portion of the study area, we found several species not found at sites to the south. We documented 13 species in and around Kugluktuk that we did not encounter elsewhere, including Anthoxanthum hirtum, Arctagrostis latifolia subsp. arundinacea (Trin.) Tzvelev, Arctous alpina (L.) Nied., Castilleja caudata (Kugluktuk and Heart Lake), Comastoma tenellum (Rottb.) Toyok., Draba nivalis Lilj., Epilobium cf. anagallidifolium, Micranthes porsildiana (Calder & Savile) Elven & D. F. Murray, Polygonum aviculare, Potamogeton gramineus, Potentilla arenosa subsp. chamissonis, Transberingia bursifolia, and Woodsia alpina. Seven species are known only from Richardson Bay at the mouth of the Richardson River, where we made collections in an extensive salt marsh habitat. These halophytic taxa include Carex ursina, Lomatogonium rotatum subsp. rotatum, Potentilla anserina subsp. groenlandica, Potentilla pulchella, Puccinellia phryganodes subsp. neoarctica (Á. Löve & D. Löve) Elven, Puccinellia vaginata (Lange) Fernald & Welsh and Suaeda calceoliformis. Other Arctic halophytes were collected at both Richardson Bay and along or near the coast of an unnamed island in the mouth of the Coppermine River, including Carex glareosa subsp. glareosa, Carex subspathacea and Gentianopsis detonsa (Rottb.) Ma subsp. detonsa. All of these taxa are likely to occur elsewhere along the mainland coast and on other islands in the mouth of the Coppermine River, where there is suitable habitat. However, none of them would be expected to occur further inland, except perhaps in areas of high salinity. For example, Suaeda calceoliformis has been recorded at inland sites on southern Victoria Island (Gillespie et al., 2015).

The study area represents an important floristic transition zone between the Subarctic and Arctic floras in the western Canadian Arctic. More than 90 species are at the northern edge of their known ranges in the study area. Some 57 taxa recorded are restricted to the mainland and not recorded from adjacent Victoria Island or any other Canadian Arctic island. These include Allium schoenoprasum, Alnus alnobetula, Anthoxanthum hirtum, Arctagrostis latifolia subsp. arundinacea, Arctostaphylos uva-ursi, Arnica frigida, Bromus pumpellianus Scribn., Calamagrostis stricta subsp. stricta, Dasiphora fruticosa, Deschampsia cespitosa (L.) P. Beauv. subsp. cespitosa, Betula occidentalis, Botrychium minganense, Botrychium neolunaria, Botrychium tunux, Carex adelostoma, Carex capitata, Carex livida, Carex podocarpa, Castilleja caudata, Castilleja raupii, Draba lonchocarpa, Draba pilosa, Eleocharis quinqueflora, Epilobium cf. anagallidifolium, Epilobium davuricum, Equisetum fluviatile, Eremogone capillaris subsp. capillaris, Erysimum coarctatum, Eurybia sibirica, Festuca altaica, Gentianopsis detonsa subsp. detonsa, Juncus alpinoarticulatus subsp. americanus, Juniperus communis subsp. depressa, Lathyrus japonicus, Linnaea borealis subsp. americana, Luzula groenlandica, Micranthes porsildiana, Petasites frigidus subsp. sagittatus, Parnassia palustris subsp. neogaea, Picea glauca, Pinguicula villosa L., Polygonum aviculare, Potamogeton gramineus, Rubus acaulis subsp. arcticus, Saussurea angustifolia (L.) DC. subsp. angustifolia, Rhododendron groenlandicum, Salix pseudomyrsinites, Salix pulchra Cham., Senecio lugens, Silene uralensis (Rupr.) Bocquet subsp. uralensis, Shepherdia canadensis, Sparganium hyperboreum Laest. ex Beurl., Stellaria borealis subsp. borealis, Stuckenia pectinata, Triglochin maritima L., Utricularia intermedia and Utricularia vulgaris.

Another 25 species at the edges of their ranges in the study area are restricted to the mainland in the western Arctic. None of these is recorded from any of the western Arctic islands, but all are known from one or more eastern Arctic islands, mostly at latitudes lower than in the study area. These include Agrostis mertensii (southern Baffin Island), Calamagrostis canadensis subsp. langsdorffii (Link) Hultén (southern Baffin Island), Calamagrostis lapponica (Wahlenb.) Hartm. (southern Baffin Island), Carex gynocrates (southern Baffin Island), Carex norvegica (southern Baffin Island and Southampton Island), Carex williamsii (southern Baffin Island and Southampton Island), Chamerion angustifolium (L.) Holub subsp. angustifolium (southern Baffin Island), Comastoma tenellum (Banks and Southampton islands), Coptidium pallasii (southern Baffin Island), Cryptogramma stelleri (southern Baffin Island), Eleocharis acicularis (southern Baffin Island), Epilobium palustre (Southampton Island), Hordeum jubatum (southern Baffin Island), Kalmia procumbens (L.) Gift & Kron ex Galasso, Banfi & F. Conti (southern Baffin Island), Lycopodium annotinum subsp. alpestre (southern Baffin Island), Pedicularis flammea (several eastern Arctic islands), Pedicularis lapponica L. (southern Baffin Island and Southampton Island), Platanthera obtusata (Banks ex Pursh) Lindl. subsp. obtusata (southern Baffin Island), Poa alpina (Baffin and Southampton islands), Primula egaliksensis Wormsk. (southern Baffin Island), Sabulina stricta (Baffin and Southampton islands), Sagina nodosa subsp. borealis (southern Baffin Island), Trichophorum cespitosum (L.) Hartm. subsp. cespitosum (southern Baffin Island), Triglochin palustris (southern Baffin Island), and Woodsia alpina (southern Baffin Island and Nottingham Island).

A third, smaller group of species at the edge of their ranges in the study area are not recorded from Victoria Island immediately adjacent to the study area, but are recorded from other islands in the western and eastern Canadian Arctic Archipelago. These include Achillea millefolium subsp. borealis (Bong.) Breitung (Banks Island and southern Baffin Island), Antennaria monocephala subsp. angustata (Greene) Hultén (numerous Arctic islands), Carex supina subsp. spaniocarpa (Steud.) Hultén (Baffin, Banks, and Southampton islands), Comastoma tenellum (Banks and Southampton islands), Draba fladnizensis Wulfen (Baffin, Banks, Cornwallis and Southampton islands), Festuca viviparoidea subsp. viviparoidea (Banks and northern Ellesmere Islands), and Pedicularis labradorica Wirsing (Banks and southern Baffin Islands). The disjunct distributions of some or all of these taxa in the Canadian Arctic Archipelago may reflect origins from different source regions. Studies of postglacial dispersal based on spatial genetic structure in widespread Arctic taxa have identified varying source areas from which taxa colonised islands in different parts of their ranges, and in some cases eastern and western Canadian Arctic populations have unique origins (Eidesen et al., 2007, 2013; Alsos et al., 2015). None of the taxa in the above lists have been characterised in this manner.

Some species that occur along the lower Coppermine River and vicinity not known from adjacent Victoria Island may yet be discovered there. Indeed, based on fieldwork on southern and northwestern Victoria Island in 2008 and 2010, we documented nine species not previously known from it or elsewhere in the western Canadian Arctic Archipelago, but known from the adjacent mainland (including the study area for most taxa). These newly recorded species were Andromeda polifolia L., Arenaria humifusa Wahl., Carex bicolor Bellardi ex All., Corallorhiza trifida Châtel. Eriophorum brachyantherum, Luzula wahlenbergii Rupr., Pinguicula vulgaris L., Sabulina stricta and Salix arctophila (Gillespie et al., 2015).

Comparisons of floristic diversity in the study area with other Arctic floras

Comparison of floristic diversity in the lower Coppermine River valley and vicinity with that in other Arctic and Subarctic regions (territorial floras, large floristic regions, local Arctic, and Subarctic floras) provides context for understanding the relative richness of the area (Table 3), although we recognise that the sizes, local environmental conditions and search effort to date in the various areas that have been studied floristically are neither necessarily equivalent nor directly comparable. As well, taxonomic concepts at family, genus and species levels vary among studies, and the historical biogeography (and therefore the floristic composition) of most areas is different.

Table 3 Family, genus, and species diversity in Arctic regions that have been studied floristically, including the study area (boldface).

Region	Ecozone	Families	Genera	Species	Taxa	Size of area (km2)	Reference	
Global Arctic	Arctic	91	420	2,218	–	71,000,000 (Walker et al., 2005)	Elven et al. (2011)	
Canadian Arctic Archipelago1 (Nunavut and Northwest Territories)	Arctic	42	141	368	375	ca. 1,424,500	Gillespie et al. (2015)	
Nunavut	Arctic and forest-tundra	65	226	6292	–	ca. 2,093,190	Brouillet et al. (2010+)3	
Northwest Territories	Arctic and forest-tundra	92	390	1,161	–	ca. 1,140,835	Brouillet et al. (2010+)3	
Lower Coppermine River valley and vicinity (Nunavut)	Arctic	45	134	300	311	ca. 1,200	Current study	
Kugluk (Bloody Falls) Territorial Park (Nunavut)	Arctic	40	103	201	207	10	Current study	
Tuktut Nogait National Park and vicinity (Northwest Territories and Nunavut)	Arctic	43	119	265	268	ca. 33,000	Saarela et al. (2013a)	
Tuktut Nogait National Park (Northwest Territories)	Arctic	35	101	215	215	ca. 18,181	Saarela et al. (2013a)	
Wager Bay region (Nunavut)	Arctic	27	65	113		ca. 19,500	Cody, Scotter & Zoltai (1989)	
Rankin Inlet (Nunavut)	Arctic	–4	–4	162	165	ca. 180	Korol (1992)	
Bathurst Inlet (Nunavut) (including Hood River)	Arctic	–4	–4	240 (270)		ca. 27,3365	Cody (1954a), Cody, Scotter & Zoltai (1984), Gould & Walker (1997), Porsild & Cody (1980)	
Chesterfield Inlet (Nunavut)	Arctic	–4	–4	160–180		–	Polunin (1940), Savile & Calder (1952)	
Nueltin Lake (Nunavut)	Forest-tundra	–4	–4	134		ca. 2646	Porsild (1950b)	
Ivvavik National Park (Yukon)	Arctic			414		10,168	Bennett (2008)	
Notes:

1 This region does not include the non-Arctic Islands in James Bay that are part of Nunavut.

2 Not including new species recorded here for Nunavut.

3 VASCAN, accessed 13 January 2016.

4 Family and genus counts not given in the relevant papers, and not scored here because genus- and family-level taxonomy has changed considerably compared to more recent studies, since 2010.

5 Area estimated from map in Cody, Scotter & Zoltai (1984).

6 Area estimated from map in Porsild (1950b).

Nunavut, Northwest Territories and the Panarctic Flora

The study area in western Nunavut includes some 70% of the family diversity, 59% of the genus diversity, and less than half (ca. 48%) of the species diversity recorded for Nunavut (excluding the species newly recorded for the territory here). The flora of Nunavut is primarily Arctic, but includes numerous boreal species, as demonstrated for the study area. Some of the boreal species in Nunavut are known only from islands in James Bay (Nunavut’s extreme southern limit) and/or the southeastern mainland. The Northwest Territories, by contrast, has a much more diverse flora compared to Nunavut. Although it includes some Arctic habitat, most of Northwest Territories comprises boreal forest and forest-tundra, with a distinct and diverse flora. Compared to the study area, Northwest Territories has twice the number of families, nearly three times the number of genera and nearly four times the number of species. All but one (Betula occidentalis) of the species recorded in the study area was included in the Panarctic flora checklist (Elven et al., 2011). The study area includes some 50% of the family diversity, 32% of the genus diversity, and 13.6% of the species diversity recorded for the Panarctic flora.

Canadian Arctic Archipelago

The Canadian Arctic Archipelago (Nunavut and Northwest Territories) is the most easily defined Arctic region in Canada, and the best known Arctic region floristically. It is phytogeographically diverse, including amphi-Beringian elements in the west and amphi-Atlantic elements in the east, in addition to widespread circumpolar and primarily North American taxa. Yet there are more plant families known in the substantially smaller study area (45 families) than in the Canadian Arctic Archipelago, where 42 families have been recorded (Aiken et al., 2007; Gillespie et al., 2015). Five families present in the study area are not recorded in the Canadian Arctic Archipelago (Amaryllidaceae, Eleagnaceae, Linnaeaceae, Ophioglossaceae, and Typhaceae); these are primarily boreal families, but each reaches the Arctic portion of the study area. Four families recorded in the Canadian Arctic Archipelago are not recorded in the study area (Crassulaceae, Diapensiaceae, Polemoniaceae, and Portulacaceae) (Aiken et al., 2007; Gillespie et al., 2015), though each is recorded elsewhere from the mainland Arctic in Canada.

Genus level diversity is slightly greater in the Canadian Arctic Archipelago (141) compared to the study area (134). Genera in the study area not known from the Canadian Arctic Archipelago include the primarily temperate- to boreal-distributed Allium L., Alnus Mill., Arctostaphylos Adans., Botrychium, Bromus L., Comarum L., Dasiphora Raf., Gentianopsis Ma, Lathyrus L., Linnaea L., Picea Mill., Populus L., Saussurea DC., and Sparganium L., while genera recorded in the Canadian Arctic Archipelago and not known in the study area include Bartsia L., Diapensia L., Diphasiastrum Holub, Euphrasia L., Montia L., Packera Á. Löve & D. Löve, Phlox L., Pleuropogon R. Br., Polemonium L., Rhodiola L., Solidago L., and Veronica L. Some of the latter genera are represented in the Canadian Arctic Archipelago by amphi-Beringian taxa known from the western Arctic islands and the western mainland Arctic (Packera hyperborealis (Greenm.) Á. Löve & D. Löve, Phlox hoodii Richardson, Polemonium boreale Adams), whereas others are amphi-Atlantic taxa known in Arctic Canada only in the eastern Canadian Arctic Archipelago and adjacent mainland (Bartsia alpina L., Euphrasia disjuncta Fernald & Wiegand, Euphrasia wettsteinii G. L. Gusarova, Diphasiastrum alpinum (L.) Holub, Rhodiola rosea L., Veronica wormskjoldii Roem. & Schult.) (Porsild & Cody, 1980; Aiken et al., 2007; Saarela et al., 2013a). Pleuropogon sabinei R. Br. is recorded from throughout the Canadian Arctic Archipelago, including adjacent Victoria Island, and along Hudson Bay on mainland Nunavut (Porsild & Cody, 1980; Cody, Scotter & Zoltai, 1989; Korol, 1992), and Montia fontana L. has a scattered boreal to low Arctic distribution (Porsild & Cody, 1980; Aiken et al., 2007). Of all these taxa, Montia fontana may be most likely to occur in the study area, given its known distribution in the Arctic. The greatest difference in diversity between the Canadian Arctic Archipelago and the study area is at the species level. There are 67 more species recorded for the Canadian Arctic Archipelago than the lower Coppermine River valley and vicinity, and there are 80 more species in the Canadian Arctic Archipelago than are recorded in the Arctic portion of the study area, but there are considerable differences in species composition between them.

On Victoria Island immediately north of the study area, some 277 taxa have been recorded (Aiken et al., 2007; Gillespie et al., 2015); further additions to the islands flora gathered by our team in 2008 and 2010 are as yet unpublished and not considered in these counts (L. J. Gillespie & J. M. Saarela, 2016, unpublished data). The large difference in species diversity between Victoria Island and the study area is consistent with the general pattern of a decrease in species diversity latitudinally from south to north, coinciding with the steep south to north temperature gradient that occurs in the Arctic over short distances (Callaghan et al., 2004). This is exemplified by differing species diversity and temperature ranges in the Arctic vegetation subzones. Victoria Island represents the southern limit of Arctic vegetation subzone D in the western Arctic—one subzone north of subzone E, which includes the Arctic portion of the study area—with a mean July temperature of 8–9 °C (vs. 9–12 °C in subzone E) and an average of 125–250 species (vs. 200–500) in local floras (Walker et al., 2005).

Local mainland Arctic floras

Floristic comparisons at smaller geographical scales indicate that the lower Coppermine River valley and vicinity is floristically rich in Nunavut relative to other regional and local areas. Vascular plant diversity has been studied in relatively few local areas on the mainland Canadian Arctic in Nunavut, Northwest Territories and Yukon, compared to local areas in the Canadian Arctic Archipelago (reviewed in Aiken et al., 2007), Arctic Quebec (Deshaye, 1983; Blondeau & Louis-Marie, 1985; Blondeau, 1989a, 1989b, 1990; Dion, Cayouette & Deshaye, 1999; Blondeau & Cayouette, 2002) and Subarctic Quebec (Payette & Lepage, 1977; Payette, Légère & Gauthier, 1978; Deshaye, Cayouette & Louis-Marie, 1988). The only local mainland Canadian Arctic area with greater known species diversity than the study area is Ivvavik National Park in northern Yukon (10,168 km2), where 414 taxa have been documented (Bennett, 2008). The diverse flora there is in part due to the treeline (and its associated boreal flora) extending very close to the Arctic coast and the unique amphi-Beringian and westerns floristic elements.

The largest mainland Arctic area in Northwest Territories and Nunavut that has been studied floristically is Tuktut Nogait National Park and vicinity, where 265 species (268 taxa) in an area of some 33,000 km2 have been recorded (Saarela et al., 2013a). The Coppermine River area, which is smaller than Tuktut Nogait National Park and vicinity, has similar diversity at the family level (43 families in Tuktut Nogait National Park and vicinity vs. 45 in the study area) and greater diversity at the genus (134 vs. 119 genera) and species (300 vs. 265 species) levels. Several species are recorded from Tuktut Nogait National Park and vicinity that are not known from the study area, including Anticlea elegans Pursh (Rydb.) var. elegans, Arctanthemum arcticum subsp. polare (Hultén) Tzvelev, Artemisia borealis subsp. richardsoniana (Besser) Korobkov, Castilleja hyperborea Pennell, Cerastium regelii Ostenf. subsp. regelii, Chrysosplenium tetrandrum Th. Fr., Draba borealis DC., Draba oligosperma Hook., Draba subcapitata Simmons, Eriophorum russeolum subsp. albidum Väre, Erigeron compositus Pursh, Gentiana prostrata Haenke, Mertensia drummondii (Lehm.) G. Don, Micranthes tenuis (Wahlenb.) Small, Packera hyperborealis (Greenm.) Á. Löve & D. Löve, Papaver cornwallisense D. Löve, Parrya arctica R. Br., Petasites frigidus (L.) Fr. subsp. frigidus, Plantago eriopoda Torr., Plantago maritima subsp. borealis (Lange) A. Blytt & O. C. Dahl, Phlox hoodii, Poa ammophila A. E. Porsild, Poa hartzii Gand. subsp. hartzii, Puccinellia angustata (R. Br.) E. L. Rand & Redfield, Puccinellia banksiensis Consaul, Puccinellia andersonii Swallen, Puccinellia tenella subsp. langeana (Berlin) Tzvelev, Puccinellia vahliana (Liebm.) Scribn. & Merr., Polemonium boreale, Potentilla hyparctica subsp. elatior (Abrom.) Elven & D. F. Murray, Potentilla subgorodkovii Jurtzev, Potentilla subvahliana Jurtzev, Ranunculus subrigidus W. B. Drew, Salix glauca var. stipulata Flod., Salix hastata L., Salix phlebophylla Andersson, Salix polaris Wahlenb., Saxifraga cespitosa L., Selaginella selaginoides (L.) P. Beauv. ex Mart. & Schrank., Silene ostenfeldii (A. E. Porsild) J. K. Morton, and Solidago multiradiata Aiton. Our study of Tuktut Nogait National Park and vicinity and the current study both include protected areas, a national park in the earlier study and a territorial park in the current one. The number of species recorded in each of these protected areas is similar, despite the protected areas being much different in size. A total of 215 species are known from Tuktut Nogait National Park (18,181 km2), whereas 208 species are known from the substantially smaller Kugluk (Bloody Falls) Territorial Park (10.5 km2). Given its very high species diversity relative to its size and compared to vascular plant diversity in the nearest Arctic national park, Kugluk (Bloody Falls) Territorial Park is clearly important from a plant and habitat conservation perspective in both the Canadian Arctic and Nunavut.

There have been three floristic studies along western Hudson Bay in Nunavut (the Subarctic flora of the Churchill region in northern Manitoba, also along Hudson Bay, is not considered here). Each of these areas is fully within the Arctic ecozone and much less diverse than the study area. The most northern of these was conducted in the Wager Bay region, an area of approximately 19,500 km2 (not including the bay), where the recorded flora is considerably less diverse (113 species) than the flora of the study area (300 species) (Cody, Scotter & Zoltai, 1989). That report is based on collections made during a survey carried out in 1984, and Cody, Scotter & Zoltai (1989) considered their species list “to be a reasonably complete list”. Most of their collection sites are now part of Ukkusiksalik National Park of Canada. This is another example of a much larger national park with much lower known vascular plant species diversity than the smaller and more diverse Kugluk/Bloody Falls Territorial Park. Further reconnaissance of the Wager Bay area, including Ukkusiksalik National Park, is likely to add considerably to the area’s species list.

In Chesterfield Inlet, south of Wager Bay, some 160–180 taxa have been recorded (Polunin, 1940; Savile & Calder, 1952). About 100 km2 south of Chesterfield Inlet, Korol (1992) recorded 162 species (165 taxa) in an area of ca. 180 km2 around Rankin Inlet, based on collections he made in 1988. Species recorded in one or more of these areas not known from the study area include Arabis arenicola (Richardson) Gelert, Astragalus eucosmus Robins., Braya purpurascens (R. Br.) Bunge, Calamagrostis deschampsioides Trin., Cochlearia officinalis L., Diapensia lapponica L., Draba lactea Adams, Luzula wahlenbergii, Micranthes tenuis, Myriophyllum exalbescens Fernald, Oxytropis bellii (Britton ex Macoun) Palib., Oxytropis hudsonica (Greene) Fernald, Phyllodoce caerulea (L.) Bab., Pleuropogon sabinei, Poa flexuosa Sm., Puccinellia tenella subsp. langeana (Berlin) Tzvelev, Salix fullertonensis C. K. Schneid., Salix fuscescens Andersson, Salix herbacea L., and Saxifraga rivularis L. Some of these are amphi-Atlantic taxa not expected in western Nunavut.

There have been two floristic studies in central mainland Nunavut. In the Bathurst Inlet region, some 240 vascular plant species have been recorded (Cody, 1954a; Porsild & Cody, 1980; Cody, Scotter & Zoltai, 1984), indicating a richer flora compared to the study sites along Hudson Bay, but a comparatively poorer flora than the study area. Several amphi-Beringian species recorded in the study area reach their western limit at Bathurst Inlet, as described in the annotated checklist. Species recorded from the Bathurst Inlet area not known from the study area include Anemone multifida Poir., Calamagrostis deschampsioides, Carex garberi Fernald, Carex tenuiflora Wahlenb., Eriophorum russeolum subsp. albidum, Luzula parviflora (Ehrh.) Desv., Myrica gale L., Oxycoccus microcarpus Turcz., Ranunculus sulphureus Sol., Salix brachycarpa Nutt., Salix fullertonensis, Salix fuscescens, and Viola pallens (Banks) Brainerd.

In a study of the Hood River, an Arctic river on mainland Nunavut that drains into Bathurst Inlet, 69–109 vascular plant species were recorded in a 17,240 km2 area along the river, with 210 species recorded in the entire study area (Gould & Walker, 1997). Details of specific sites where species have been recorded along the Hood River are not given in Gould & Walker (1997) and have neither been published elsewhere nor is it clear from their paper whether or not all of the species they recorded are vouchered with specimens (housed in the University of Colorado Herbarium (COLO) and ALA). The known flora of the Hood River is less rich than that of the lower Coppermine River, with nearly 100 fewer recorded species. However, the relationship between species richness and the local environment, and the structure and diversity of the vegetation, are much better characterised for the Hood River than the Coppermine River (Gould & Walker, 1997, 1999). The biodiversity data in the current study would serve as an excellent baseline for a more detailed characterisation of the vegetation along the Coppermine River, taking an ecological approach similar to that used along the Hood River. Vascular plants have also been studied in the Yathkyed Lake basin and along the Kazan River in interior Nunavut. Alfred E. Porsild collected some 500 numbers in these areas in 1930 (Porsild, 1950b). The collections were mapped in Porsild & Cody (1980), but never published in detail, so floristic comparisons with this part of mainland Nunavut are difficult.

There has been only one previous floristic study of an area within the forest-tundra in Nunavut. Porsild (1950b) reported on a collection of some 300 numbers of vascular plants made by zoologist Francis Harper in 1947 around northwestern Nueltin Lake in southeastern Nunavut. A total of 80–90% of the land is tundra in that area, with stands of black spruce and tamarack in the muskegs and on upland slopes (Porsild, 1950b). The collections represented 134 species, including numerous forest species not recorded elsewhere on mainland Nunavut, including the study area, such as Agrostis scabra Willd., Betula papyrifera Marsh, Carex canescens L., Carex disperma Dewey, Carex leptalea Wahlenb., Carex limosa L., Diphasiastrum complanatum (L.) Holub, Equisetum sylvaticum L., Erigeron acris var. kamtschaticus (DC.) Herder, Erigeron elatus (Hook.) Greene, Geocaulon lividum (Richardson) Fernald, Gymnocarpium robertianum (Hoffm.) Newman, Larix laricina (Du Roi) Koch, Picea mariana (Mill.) B. S. P., Potamogeton friesii Rupr., Potamogeton alpinus Balb., Potamogeton praelongus Wulfen, Mitella nuda L., Ribes triste Pall., Rubus strigosus Michx., Pyrola minor L., Menyanthes trifoliata L., Galium labradoricum (Wiegand) Wiegand and Viburnum edule (Michx.) Raf. Based on knowledge of the Nueltin Lake area flora and surrounding areas, Porsild (1950b) estimated there to be at least 300 species in the forested portion of what is now southern Nunavut, adjacent to the southern limit of the Arctic. Floristic diversity in that little explored area remains poorly understood.

The relatively high species diversity in the study area can be attributed to (1) the large number of boreal species recorded in the area, many of which extend into the Arctic ecozone along the Coppermine River valley associated with the northern extent of the forest-tundra, and some of which may be relicts of a post-glaciation warmer period; (2) varying topography in the area, providing numerous microhabitats for species; (3) varying geology, including a mix of calcareous and acidic substrates, each of which support different species; (4) and considerable search effort in the area over time. Callaghan et al. (2004) noted about 40% of the global Arctic vascular plant flora is represented by boreal species that barely penetrate the Arctic, occurring close to the treeline and/or along large rivers connecting the Subarctic and the Arctic. This is indeed the general distribution pattern of the boreal element of the lower Coppermine River flora. Callaghan et al. (2004) suggested that the boreal species that already extend into the Arctic are likely to be the ones that may colonise further with continued warming. The northern range extensions, we have documented for boreal taxa along the Coppermine River, however, are likely long-established species that simply have not previously been recorded, rather than recent colonisers. Accordingly, the floristic data presented here should serve as a useful baseline for understanding potential future change in the distributions of these species in the study area, and for documenting the distributions of species that may eventually colonise the study area.

Organisms at the edge of their range are of considerable interest to ecologists and evolutionary biologists (Hardie & Hutchings, 2010), and genetic characterisation of such populations may improve understanding of their adaptations to their local environments (e.g., Olson et al., 2013), which in turn may improve models predicting how species may respond to climate change (Wullschleger et al., 2015). Known sites of many of the species at their northern limits in the study area, particularly in and around Kugluktuk, and Kugluk (Bloody Falls) Territorial Park, are readily accessible and these would be ideal places for further genetic sampling to address such questions.

Annotated checklist of vascular plants

Lycophytes

Lycopodiaceae [2/2]

Huperzia arctica (Tolm.) Sipliv.—Arctic fir clubmoss | Circumpolar?

Previously recorded from Kugluktuk (Porsild & Cody, 1980). We made collections at Fockler Creek, Kugluktuk, Heart Lake and Kugluk (Bloody Falls) Territorial Park. Widespread throughout the Canadian Arctic (Cody, Scotter & Zoltai, 1989; Korol, 1992; Aiken et al., 2007; Dignard, 2013d; Saarela et al., 2013a; Bennett, 2015). Recognition of Huperzia arctica in the Canadian Arctic as distinct from the boreal Huperzia selago (L.) Bernh. ex. Schrank & Mart. follows Elven et al. (2011). The taxonomic history of the complex is reviewed by Jukonienė, Dobravolskaitė & Sendžikaitė (2012). Wagner & Beitel (1993a) did not map the taxon, recognised as Huperzia selago, as occurring in the Canadian Arctic, despite numerous records from the mainland and Arctic islands of Nunavut, most known prior to their publication (Porsild & Cody, 1980, as Lycopodium selago L.; Cody & Britton, 1989, as Huperzia selago subsp. selago; Aiken et al., 2007, as Huperzia selago; Saarela et al., 2013a).

Specimens Examined: Canada. Nunavut: Kitikmeot Region: Coppermine [Kugluktuk], Coronation Gulf, at mouth of Coppermine River [67.822146°N, 115.078387°W ± 0.5 km], 4 August 1948, H. T. Shacklette 3281 (CAN-200060); Kugluktuk, flat mesa at top of North Hill, 67°49′32″N, 115°6′39″W ± 100 m, 50 m, 29 June 2014, Saarela, Sokoloff & Bull 3089 (CAN); E side of Fockler Creek, in valley just above creek’s confluence with the Coppermine River, ca. 1.4 km south-southwest (SSW) of Sandstone Rapids, 67°26′14.5″N, 115°38′34.8″W ± 50 m, 146 m, 4 July 2014, Saarela, Sokoloff & Bull 3348 (CAN); Kugluk (Bloody Falls) Territorial Park, slope above Bloody Falls (W side) where Coppermine River narrows to Bloody Falls, 67°44′23″N, 115°22′35.9″W ± 25 m, 33 m, 12 July 2014, Saarela, Sokoloff & Bull 3849 (CAN); ca. 0.5 km SW of Heart Lake, SW of Kugluktuk, 7.5 km SW of mouth of Coppermine River, 67°47′52″N, 115°14′14.4″W ± 350 m, 66 m, 23 July 2014, Saarela, Sokoloff & Bull 4340 (CAN).

Lycopodium annotinum subsp. alpestre (Hartm.) Á. Löve & D. Löve—Stiff clubmoss | Circumpolar-alpine | Noteworthy Record

Newly recorded for the study area, where we found a single large patch in a Subarctic snowbed community below a southeast-facing ridge near Tundra Lake, growing with Trichophorum cespitosum, Salix spp. and Anemone richardsonii. Our collection closes a distribution gap between Great Bear Lake and Bathurst Inlet, and represents the species’ known northern limit in the Central Arctic/Subarctic. Elsewhere on mainland Nunavut recorded from the Bathurst Inlet area, where known from at least three collections (Porsild & Cody, 1980), including Kelsall & McEwen 274 (CAN-202945) and 7 (CAN-202944); Hood River (Gould & Walker, 1997); sites just north of the treeline along the Nunavut/Northwest Territories border (64°25′N, 108°54′W, 1938, Carroll s.n., CAN-211367; Contwoyto Lake, 65°42′N, 110°50′W, 1938, Carroll s.n. (CAN-211368)—neither mapped in Porsild & Cody (1980)); Thelon River ca. 28 miles SW of Beverly Lake, Kuyt 22 (CAN-561541); southeastern Nunavut, including Edlund 523 (CAN-502409), Boles et al. RB00-22 (CAN-592207) and RB00-14 (CAN-592205); and other sites (Gussow, 1933; Porsild, 1950b) mapped in Porsild & Cody (1980). A collection bearing only “Arctic North America. Dr. Richardson, CAN-4746” may be from the study area. In the Canadian Arctic Archipelago known only from southern Baffin Island (Aiken et al., 2007) and recorded in adjacent northern Quebec and Labrador (Dignard, 2013d). We follow Elven et al. (2011) in recognising the northern alpine-boreal to southern Arctic plants as subsp. alpestre. Porsild & Cody (1980) treated the taxon as Lycopodium annotinum s.l., while Wagner & Beitel (1993b) did not recognise infraspecific taxa and considered previously recognised taxa as environmentally induced forms.

Specimens Examined: Canada. Nunavut: Kitikmeot Region: meadow just S of Tundra Lake, ca. 4.2 km SE of Sandstone Rapids, Coppermine River, 67°25′29.5″N, 115°33′50.4″W ± 50 m, 266 m, 5 July 2014, Saarela, Sokoloff & Bull 3430 (CAN, UBC).

Monilophytes

Cystopteridaceae [1/1]

Recognition of Cystopteridaceae (Payer) Shmakov as a family separate from Woodsiaceae, in which Cystopteris was recently treated (Smith et al., 2006), follows Christenhusz, Zhang & Schneider (2011) and Rothfels et al. (2012).

Cystopteris fragilis (L.) Bernh., Fig. S2A—Fragile fern, brittle fern | Cosmopolitan

Previously recorded from Kugluktuk (Porsild & Cody, 1980). We made collections at Kendall River, Kugluk (Bloody Falls) Territorial Park and Kugluktuk. Widespread throughout the Canadian Arctic (Porsild & Cody, 1980; Cody, Scotter & Zoltai, 1989; Korol, 1992; Cody, Reading & Line, 2003; Cody & Reading, 2005; Aiken et al., 2007; Dignard, 2013a; Saarela et al., 2013a).

Specimens Examined: Canada. Nunavut: Kitikmeot Region: Coppermine [Kugluktuk], on ledge overlooking Coppermine River near settlement [67.826667°N, 115.09333°W ± 1.5 km], 11 July 1958, R. D. Wood s.n. (CAN-265613); confluence of Coppermine and Kendall rivers (NW side of Coppermine River, S side of Kendall River), 67°6′51.1″N, 116°8′18.3″W ± 150 m, 220 m, 7 July 2014, Saarela, Sokoloff & Bull 3577 (CAN, UBC); Kugluk (Bloody Falls) Territorial Park, rocky valley immediately SW of Bloody Falls, along rough marked section of Portage Trail, 67°44′34″N, 115°22′16″W ± 50 m, 20 m, 13 July 2014, Saarela, Sokoloff & Bull 3887 (CAN, US); Kugluk (Bloody Falls) Territorial Park, S-facing cliff (gabbro sill) above start of Bloody Falls, W side of Coppermine River, W side of Portage Trail, 67°44′23.2″N, 115°22′54.5″W ± 50 m, 57 m, 16 July 2014, Saarela, Sokoloff & Bull 4075 (CAN); SE edge of Kugluktuk, rocky cliffs overlooking Coppermine River, 67°49′9.2″N, 115°5′40.4″W ± 50 m, 28 m, 24 July 2014, Saarela, Sokoloff & Bull 4356 (ALA, CAN).

Dryopteridaceae [1/1]

Dryopteris fragrans (L.) Schott, Figs. S2B and S2C—Fragrant wood fern, fragrant shield fern | European (NE)–Asian–amphi-Beringian–North American (N)

Previously reported from Kugluktuk (Cody, 1954b; Porsild & Cody, 1980; Cody & Britton, 1989). We made collections at Kugluktuk, Tundra Lake, Bigtree River and Kugluk (Bloody Falls) Territorial Park. Elsewhere in the Canadian Arctic recorded from Banks, Baffin, Ellesmere, Southampton and Victoria islands, and numerous mainland sites (Porsild & Cody, 1980; Cody, Scotter & Zoltai, 1989; Korol, 1992; Cody & Reading, 2005; Aiken et al., 2007; Dignard, 2013b; Saarela et al., 2013a; Bennett, 2015).

Specimens Examined: Canada. Nunavut: Kitikmeot Region: Coppermine [Kugluktuk] [67.8333°N, 115.1°W], 5 July 1958, R. D. Wood s.n. (CAN-265611); Coppermine [Kugluktuk], 67°49′36″N, 115°5′36″W, 1 August 1951, W. I. Findlay 227 (DAO-171347 01-01000617086, MT00194660); Kugluktuk, flat mesa at top of North Hill, 67°49′32″N, 115°6′39″W ± 100 m, 50 m, 29 June 2014, Saarela, Sokoloff & Bull 3091 (CAN, MO, O); meadow just S of Tundra Lake, ca. 4.2 km SE of Sandstone Rapids, Coppermine River, 67°25′29.5″N, 115°33′50.4″W ± 50 m, 266 m, 5 July 2014, Saarela, Sokoloff & Bull 3429 (CAN, MT, UBC); meadow just S of Tundra Lake, ca. 4.2 km SE of Sandstone Rapids, Coppermine River, 67°25′34.8″N, 115°33′27.8″W ± 20 m, 265 m, 5 July 2014, Saarela, Sokoloff & Bull 3434 (CAN); confluence of Coppermine and Bigtree rivers, 66°56′23.8″N, 116°21′3.2″W ± 100 m, 265 m, 7 July 2014, Saarela, Sokoloff & Bull 3603 (CAN, US); S-facing slopes on W side of Coppermine River, about halfway between Escape Rapids and Muskox Rapids, 67°31′18.2″N, 115°36′20.1″W ± 150 m, 115 m, 8 July 2014, Saarela, Sokoloff & Bull 3629 (ALA, CAN); Kugluk (Bloody Falls) Territorial Park, upper ledges of rocky (gabbro) S-facing cliffs above the start of Bloody Falls (W bank of River), just E of Portage Trail, 67°44′21.7″N, 115°22′42.2″W ± 25 m, 46 m, 14 July 2014, Saarela, Sokoloff & Bull 3934 (ALTA, CAN).

Equisetaceae [1/5]

Equisetum arvense subsp. alpestre (Wahlenb.) Schönswetter & Elven, Fig. 13A—Alpine field horsetail | Circumpolar-alpine

Figure 13 Equisetum arvense subsp. alpestre, Equisetum fluviatile and Equisetum palustre.

Equisetum arvense subsp. alpestre: (A) habit, vicinity of lower Coppermine River, Nunavut, 67°45′2.41″N, 115°22′22.29″W, 17 July 2014. Equisetum fluviatile: (B) habit, Saarela et al. 3878. (C) Habitat, Saarela et al. 3878. Equisetum palustre: (D) habitat, vicinity of Bloody Falls, Kugluk (Bloody Falls) (Bloody Falls) Territorial Park, Nunavut, 13 July 2014. Photographs by R. D. Bull (A, D) and J. M. Saarela (B, C).

Previously recorded from Kugluktuk (Cody, 1954b; Porsild & Cody, 1980). We made collections at Fockler Creek and Kugluk (Bloody Falls) Territorial Park. Taxonomy follows Elven et al. (2011), who recognise northern plants as subsp. alpestre and more southerly ones as subsp. arvense. However, according to the maps in Aiken et al. (2007) the two subspecies are sympatric in the Canadian Arctic Archipelago. Their distributions have not been mapped for the rest of North America. Mainland collections from Tuktut Nogait National Park and vicinity are subsp. alpestre (Saarela et al., 2013a). Equisetum arvense s.l. is widespread across the Arctic mainland (Porsild & Cody, 1980; Korol, 1992; Cody, Reading & Line, 2003; Dignard, 2013c; Bennett, 2015).

Specimens Examined: Canada. Nunavut: Kitikmeot Region: Coppermine [Kugluktuk], 67°49′36″N, 115°5′36″W, 6 August 1951, W. I. Findlay 260 (DAO-171039 01-01000617100, QFA0229749); Coppermine [Kugluktuk], 67°51′N, 115°16′W, 2 July 1972, F. Fodor N 161 (UBC-V151902); Kugluktuk, 67.825517°N, 115.09015°W, 24 July 2006, J. Davis 640 (CAN-597647); spruce forest along Fockler Creek, ca. 2.3 km south-southeast (SSE) of Sandstone Rapids, Coppermine River, 67°25′45.7″N, 115°37′21.8″W ± 25 m, 166 m, 2 July 2014, Saarela, Sokoloff & Bull 3203 (CAN); S of Fockler Creek, along small tributary that runs into Fockler Creek, ca. 2.3 km S of Sandstone Rapids, Coppermine River, 67°25′44.9″N, 115°38′25.9″W ± 100 m, 152 m, 3 July 2014, Saarela, Sokoloff & Bull 3272 (CAN); Kugluk (Bloody Falls) Territorial Park, flats above boardwalk W of Bloody Falls, 67°44′34.5″N, 115°22′27″W ± 100 m, 135 m, 16 July 2014, Saarela, Sokoloff & Bull 4057 (CAN, UBC).

Equisetum fluviatile L., Figs. 13B and 13C—Water horsetail | Circumboreal | Noteworthy Record

Newly recorded for the study area and western Nunavut, and a range extension from the nearest known sites along the eastern arm of Great Bear Lake (Porsild & Cody, 1980; Cody & Britton, 1989). This distinctive horsetail species is rare in the study area. We encountered two small populations. One was found along the edge of a small sheltered pond in Kugluk (Bloody Falls) Territorial Park with Utricularia vulgaris, and the other along the edge of a small body of water with Carex aquatilis subsp. stans at the edge of a sand bar in the floodplain of the Coppermine River just south of Kugluktuk. Both sites are within the Arctic ecozone, and are the northernmost sites for the taxon in Nunavut. Elsewhere in Arctic Nunavut known from the Belcher Islands (Sanikiluaq, Consaul et al. 3875, CAN-599928) (Dignard, 2013c) and two mainland collections: Thelon River ca. 28 mi SW of Beverly Lake, 64°36′N, 100°30′W, 11 July 1960, E. Kuyt 62, CAN-561540, mapped in Porsild & Cody (1980); and east of South Bay of Griffin Lake, Reading s.n., DAO [east side of Queen Maud Bird Sanctuary] (Cody, 1996a; Dignard, 2013c). There are non-Arctic Nunavut records from adjacent to the Manitoba border (Nueltin Lake, Harper 2378, CAN-3849) (Porsild & Cody, 1980; Hauke, 1993) (not mapped in Cody & Britton (1989)) and Akimiski Island (Blaney & Kotanen, 2001). It also reaches the Arctic in the Anderson River Delta (Northwest Territories: Grizzly Bear Creek, 69°42′N, 129°W, 24 August 1959, T. D. Barry 457, CAN-287853), on the north slope of Alaska (specimens at ALA; Porsild & Cody, 1980) and in northern Labrador (Dignard, 2013c).

Specimens Examined: Canada. Nunavut: Kitikmeot Region: Kugluk (Bloody Falls) Territorial Park, rocky valley immediately SW of Bloody Falls, along rough marked section of Portage Trail, 67°44′34″N, 115°22′16″W ± 50 m, 20 m, 13 July 2014, Saarela, Sokoloff & Bull 3878 (CAN, UBC); flats below large overhanging cliffs above Coppermine River, just S of Kugluktuk, 67°48′56.7″N, 115°6′22.6″W ± 10 m, 2 m, 26 July 2014, Saarela, Sokoloff & Bull 4409 (ALA, CAN).

Equisetum palustre L., Fig. 13D—Marsh horsetail | Circumboreal | Noteworthy Record

Newly recorded for the study area. Not recorded for Nunavut by Porsild & Cody (1980), Cody & Britton (1989) or Hauke (1993), but more recently recorded for the territory from the southeastern mainland (Reading s.n. in 1981 and 1982, DAO, Cody, Reading & Line, 2003), west of the Coppermine River (Reading 8, 9, 47, DAO), Bathurst Inlet (Reading 204, DAO), and west of Arviat (Reading 121, DAO) (Cody & Reading, 2005). We made collections at Fockler Creek and Kugluk (Bloody Falls) Territorial Park, which represent a minor northeastern range extension with respect to Reading’s collections from just west of the study area. The next-nearest records are from the Great Bear Lake area (Porsild, 1943; Porsild & Cody, 1980). This species forms extensive, bright green stands in mesic tundra along the north side of the Portage Trail in Kugluk (Bloody Falls) Territorial Park. Elsewhere in the Canadian Arctic recorded from Tuktut Nogait National Park and vicinity (Saarela et al., 2013a) and northern Labrador (Dignard, 2013c).

Specimens Examined: Canada. Nunavut: Kitikmeot Region: S of Fockler Creek, along small tributary that runs into Fockler Creek, ca. 2.3 km S of Sandstone Rapids, Coppermine River, 67°25′44.9″N, 115°38′25.9″W ± 100 m, 152 m, 3 July 2014, Saarela, Sokoloff & Bull 3244 (CAN); S of Fockler Creek, along small tributary that runs into Fockler Creek, ca. 2.3 km S of Sandstone Rapids, Coppermine River, 67°25′44.9″N, 115°38′25.9″W ± 100 m, 152 m, 3 July 2014, Saarela, Sokoloff & Bull 3253 (CAN, UBC); SE-facing slopes above Escape Rapids, W side of Coppermine River, 67°36′49.8″N, 115°29′27.4″W ± 10 m, 67 m, 8 July 2014, Saarela, Sokoloff & Bull 3741 (ALA, CAN); Kugluk (Bloody Falls) Territorial Park, rocky valley immediately SW of Bloody Falls, along rough marked section of Portage Trail, head of small unnamed pond just W of falls, 67°44′42.8″N, 115°22′29.2″W ± 10 m, 9 m, 13 July 2014, Saarela, Sokoloff & Bull 3904 (ALTA, CAN); Kugluk (Bloody Falls) Territorial Park, W side of Coppermine River, between Sandy Hills and Bloody Falls, 67°45′13.2″N, 115°22′6.3″W ± 3 m, 21 m, 17 July 2014, Saarela, Sokoloff & Bull 4143 (CAN, O).

Equisetum scirpoides Michx.—Dwarf scouring rush | Circumboreal–polar | Noteworthy Record

Our collections, from Fockler Creek, Kendall River, and Kugluk (Bloody Falls) Territorial Park, are the first ones for the study area and extend the range of the species north in this area, from the Big Bend Area of the Coppermine River just south of the study area (Reading 28, 39-3, DAO; Cody, Reading & Line, 2003). Elsewhere on mainland Nunavut there are at least five records (Porsild & Cody, 1980; Cody, Scotter & Zoltai, 1984; Cody & Britton, 1989; Cody, Scotter & Zoltai, 1989), with the nearest collection to the east from the Bathurst Inlet region; the shaded range map in Hauke (1993) covers nearly all of mainland Nunavut. To the west it reaches the Arctic on mainland Northwest Territories, Yukon and Alaska (Porsild & Cody, 1980; Cody, Scotter & Zoltai, 1992; Saarela et al., 2013a). In the Canadian Arctic Archipelago recorded from southern Baffin Island, Banks, Coats, King William, Southampton and Victoria islands (Aiken et al., 2007; L. J. Gillespie & J. M. Saarela, 2016, unpublished data), and also recorded in Arctic Quebec (Dignard, 2013c).

Specimens Examined: Canada. Nunavut: Kitikmeot Region: sedge meadow adjacent to small lake on flats N of Fockler Creek, ca. 1.5 km SSE of Sandstone Rapids, Coppermine River, 67°26′8.8″N, 115°37′35.9″W ± 20 m, 168 m, 2 July 2014, Saarela, Sokoloff & Bull 3226 (CAN); N side of Fockler Creek, ca. 1.9 km S of Sandstone Rapids, Coppermine River, 67°25′57.89″N, 115°38′3.9″W ± 10 m, 162 m, 4 July 2014, Saarela, Sokoloff & Bull 3320 (CAN, UBC); E side of Fockler Creek, in valley just above creek’s confluence with the Coppermine River, ca. 1.4 km SSW of Sandstone Rapids, 67°26′14.5″N, 115°38′34.8″W ± 50 m, 146 m, 4 July 2014, Saarela, Sokoloff & Bull 3350 (ALA, CAN); confluence of Coppermine and Kendall rivers (NW side of Coppermine River, S side of Kendall River), 67°6′51.1″N, 116°8′18.3″W ± 150 m, 220 m, 7 July 2014, Saarela, Sokoloff & Bull 3582 (ALTA, CAN); Kugluk (Bloody Falls) Territorial Park, SE-facing slope above small stream in deep gully that runs into Coppermine River just below Bloody Falls, ca. 1 km W of Bloody Falls, 67°44′41.2″N, 115°23′34.8″W ± 50 m, 49 m, 15 July 2014, Saarela, Sokoloff & Bull 4033 (CAN).

Equisetum variegatum Schleich. subsp. variegatum—Variegated scouring rush, variegated horsetail | Circumpolar-alpine

Previously recorded from the study area (Porsild & Cody, 1980), but we were unable to locate a voucher specimen. We made collections at Fockler Creek, Bigtree River, and Kugluk (Bloody Falls) Territorial Park. Recently recorded from the nearby Big Bend Area of the Coppermine River (Reading 29, DAO, Cody, Reading & Line, 2003) and widespread throughout the Canadian Arctic (Porsild & Cody, 1980; Korol, 1992; Aiken et al., 2007; Dignard, 2013c; Saarela et al., 2013a).

Specimens Examined: Canada. Nunavut: Kitikmeot Region: old riverbed of Fockler Creek, ca. 2.3 km SSE of Sandstone Rapids, Coppermine River, 67°25′48″N, 115°37′33″W ± 25 m, 153 m, 1 July 2014, Saarela, Sokoloff & Bull 3151 (CAN, UBC); S of Fockler Creek, along small tributary that runs into Fockler Creek, ca. 2.3 km S of Sandstone Rapids, Coppermine River, 67°25′44.9″N, 115°38′25.9″W ± 100 m, 152 m, 3 July 2014, Saarela, Sokoloff & Bull 3256 (CAN); small unnamed tributary of Sleigh Creek, just upstream from its confluence with Sleigh Creek, 67°26′43.9″N, 115°36′57″W ± 5 m, 156 m, 6 July 2014, Saarela, Sokoloff & Bull 3464 (ALA, CAN); confluence of Coppermine and Bigtree rivers, 66°56′23.8″N, 116°21′3.2″W ± 100 m, 265 m, 7 July 2014, Saarela, Sokoloff & Bull 3602 (CAN); Kugluk (Bloody Falls) Territorial Park, rocky beach above Bloody Falls, W bank of Coppermine River, 67°44′18″N, 115°22′57.3″W ± 250 m, 34 m, 14 July 2014, Saarela, Sokoloff & Bull 3976 (ALTA, CAN).

Ophioglossaceae [1/3]

Botrychium Sw.

The wide-ranging taxon Botrychium lunaria has recently been segregated into multiple taxa based on genetic and morphological data. In the complex, two new species were described in 2002 (Botrychium tunux M. Stensvold & D. Farrar; Botrychium yaaxudakeit M. Stensvold & D. Farrar) and three new taxa were recently described (Botrychium lunaria var. melzeri Stensvold & Farrar, Botrychium neolunaria Stensvold & Farrar and Botrychium nordicum Stensvold & Farrar) (Stensvold, Farrar & Johnson-Groh, 2002; Stensvold & Farrar, 2008; Stensvold, 2007; Stensvold & Farrar, 2016). Dauphin, Vieu & Grant (2014) sampled multiple accessions of the Botrychium lunaria complex for three plastid regions, and found support for recognition of the above-named taxa. They found Botrychium minganense to be a subclade that is part of a broader lineage referred to as the Simplex-Campestre clade, with 14 species, and the Simplex-Campestre clade and the Lunaria clade, the latter comprising the Botrychium lunaria complex, to be sister taxa. They suggested the psbA-trnH intergenic spacer region is the most useful of the plastid regions studied to serve as a DNA barcode for the complex.

Because we had difficulties identifying our Botrychium collections based on morphology alone, we collected DNA barcode data for our samples to aid in identification. We amplified and sequenced psbA-trnH and rbcL for three individuals from our new collections, here divided into four separate numbers on the basis of morphological variation (3755a, 3755b, 4179a, and 4179b), and for the two collections (Gillespie et al. 9209, 9281) from adjacent Northwest Territories we previously identified as Botrychium lunaria (Saarela et al., 2013a). GenBank accession numbers for our new sequences are KT906414–KT906423. We sequenced psbA-trnH to compare sequences of our collections to those for the Botrychium lunaria complex and Botrychium minganense generated by Dauphin, Vieu & Grant (2014) (GenBank accession numbers of their sequences in the trees are prefixed by “KF”) and one other available sequence (AB575330: Ebihara, Nitta & Ito, 2010). Dauphin, Vieu & Grant (2014) did not obtain psbA-trnH for the eight other species in their Simplex-Campestre clade. We sequenced rbcL, one of the two core barcode loci for vascular plants, to compare our samples with existing rbcL sequences for Botrychium lunaria (here referred to as Botrychium lunaria s.l., recognising that the recent taxonomic changes were not taken into account in earlier works), Botrychium minganense and related taxa (L40968–L40972, L40981: Hauk, 1995; AB574664: Ebihara, Nitta & Ito, 2010; DQ849133, DQ849172, DQ849169, DQ849146: Hauk, Kennedy & Hawke, 2012; JN965301–JN965304: Kuzmina et al., 2012; KC482120-KC482: Saarela et al., 2013b; AB626656: Shinohara et al., 2013). For each analysis we designated Botrychium boreale J. Millde as an outgroup taxon, following recent phylogenies (Hauk, Kennedy & Hawke, 2012; Dauphin, Vieu & Grant, 2014). DNA was extracted from silica gel dried leaf material using a silica-based column method (Alexander et al., 2007) similar to commercially available DNA extraction kits. psbA-trnH was amplified using primers psbAF (Sang, Crawford & Stuessy, 1997) and trnH2 (Tate & Simpson, 2003). The rbcLa region was amplified using primers rbcLa-F (primer P1630 in Levin et al., 2003) and rbcLa-R (primer SI_Rev in Kress et al., 2009). Amplification was performed in a 15 μL volume with a final concentration of 1x HF Phusion buffer, 1.5 mmol/L MgCl2, 0.2 mmol/L dNTP, 0.5 μmol/L of each primer, 0.3 units of Phusion polymerase, and 1.5 μL of extracted DNA. The following cycling conditions were used: 30 s at 98 °C, followed by 34 cycles of 98 °C for 10 s, 60 °C (psbA-trnH) or 55 °C (rbcL) for 30 s, and 72 °C for 30 s, followed by a final extension step at 72 °C for 5 min. Amplification products were purified using the enzymes exonuclease I and shrimp alkaline phosphatase. Big Dye version 3.1 (Life Technologies Corporation, Carlsbad, CA, USA) was used for sequencing reactions using 0.4 μL of Big Dye in a 10 μL reaction. Sequencing reaction products were purified via ethanol—EDTA—sodium acetate precipitation. Nucleotide sequences were generated using automated cycle-sequencing on an Applied Biosystems 3130xl automated sequencer. Using Geneious 8.0.4, we aligned our new sequences with the sequences obtained from GenBank and, for each alignment, we constructed neighbour joining trees under a Jukes–Cantor substitution model. Results are described in the species accounts below.

Botrychium minganense Vict., Fig. 14—Mingan moonwort | Circumboreal | Noteworthy Record

Figure 14 Botrychium minganense.

(A) Habit, Saarela et al. 4179a. (B) Habitat, Saarela et al. 4179a. Photographs by R. D. Bull.

This is a widespread allotetraploid taxon variously recognised as a species (Victorin, 1927; Cody & Britton, 1989; Wagner & Wagner, 1993) or as Botrychium lunaria var. minganense (Vict.) Dole (Porsild, 1951, 1966, 1975; Porsild & Cody, 1980). Species recognition is supported by morphological and genetic data (Wagner & Lord, 1956; Stensvold, 2007; Dauphin, Vieu & Grant, 2014). Distinguishing morphological characters are summarised in the key below. In the neighbour-joining analyses, the plastid data place one of our collections (4179a) in a cluster with individuals of Botrychium minganense and related taxa, although our sequences are not identical to individuals identified as Botrychium minganenese. The psbA-trnH sequence is part of a cluster with the Botrychium minganense accessions, which are all Alaskan plants (Dauphin, Vieu & Grant, 2014) (Fig. 15). Our psbA-trnH sequence differs from the others of Botrychium minganense by four substitutions. Further sampling is needed to determine the geographic distribution of the Coppermine River psbA-trnH haplotype. The rbcL sequence of this sample is part of a cluster comprising Botrychium gallicomontanum Farrar & Johnson-Groh, Botrychium minganense, Botrychium pallidum W. H. Wagner, Botrychium mormo W. H. Wagner, Botrychium paradoxum W. H. Wagner, and Botrychium watertonense W. H. Wagner. Our sequence is identical to those of Botrychium pallidum (scattered distribution in central North America), Botrychium mormo (endemic to Michigan, Minnesota and Wisconsin) and Botrychium gallicomontanum (endemic to Minnesota) (Wagner & Wagner, 1993), which other studies have found to be closely related to Botrychium minganense (Hauk, Kennedy & Hawke, 2012; Dauphin, Vieu & Grant, 2014). It differs from the four published sequences of Botrychium minganense and those of Botrychium paradoxum and Botrychium watertonense by two one-bp indels and one substitution. These six published sequences were obtained in 1995 or before using early sequencing technology (autoradiography) (Hauk, 1995); sequences for these taxa should be obtained using contemporary methods to ensure the variation in the earlier sequences is not artefactual. The plants of collection 4179a correspond morphologically to Botrychium minganense and, given their placement in the genetic trees, we place them under this species. Our collection is the first record for the study area. The plants grew in moist (seasonally wet) sand with Botrychium tunux (no. 4179b), Castilleja caudata, Juncus arcticus subsp. alaskanus, Lupinus arcticus, Parnassia kotzebuei, Pinguicula vulgaris, and Salix alaxensis var. alaxensis.

Figure 15 Neighbour-joining analyses (Jukes–Cantor substitution model) of new and previously published psbA-trnH and rbcL DNA sequences of Botrychium.

Newly sequenced individuals are in boldface. (A) psbA-trnH dendrogram. (B) rbcL dendrogram.

Cody, Reading & Line (2003) reported three Botrychium lunaria collections gathered in 1999 from the nearby Big Bend area of the Coppermine River (Reading 21, DAO-784829 01-01000677959; Reading 37-1, DAO-7848333 01-01000677496; Reading s.n., DAO-731189 01-01000677960). These specimens were re-determined to Botrychium minganense by L. Gillespie in 2012 (Saarela et al., 2013a) and are the first confirmed records for mainland Nunavut. No genetic data from these collections have been obtained. An unpublished collection from north of Dismal Lakes and west of the Coppermine River (Reading 64, DAO) is a mixed collection, comprising plants of Botrychium minganense and Botrychium tunux. The presence of Botrychium minganense in the Coppermine River valley and vicinity is well beyond the range given in Wagner & Wagner (1993), which in the North included southern Yukon and a small portion of southwestern Northwest Territories. Bennett et al. (2010 [2011]) reported it to be widespread in Yukon. Elsewhere in Nunavut recorded from Akimiski Island (Blaney & Kotanen, 2001). No Botrychium taxa are recorded for the Canadian Arctic Archipelago (Aiken et al., 2007), and there is a single Arctic record from northern Quebec (Dignard, 2013e).

Specimens Examined: Canada. Nunavut: Kitikmeot region: shallow gully in sand hills above Bloody Falls, SE side of Coppermine River on SW-facing slopes, 67°44′28.2″N, 115°22′3″W ± 15 m, 78 m, 19 July 2014, Saarela, Sokoloff & Bull 4179a (CAN).

Botrychium neolunaria Stensvold & Farrar—New World moonwort | Noteworthy Record

Some North American plants previously treated as Botrychium lunaria are now recognised as a separate species on the basis of genetic and morphological data (Stensvold, 2007). This taxon was referred to as Botrychium neolunaria ined. by Stensvold (2007). Although the species name was not validly published until recently (Stensvold & Farrar, 2016), it is in use in the literature (Nawrocki, Fulkerson & Carlson, 2013; Dauphin, Vieu & Grant, 2014). Diagnostic characters are listed by Farrar (2011), who noted the taxon cannot be reliably distinguished morphologically from Botrychium lunaria var. lunaria, and their ranges overlap in North America only in high mountain habitats in Alaska and Yukon, and in eastern Canada. Dauphin, Vieu & Grant (2014) noted them to be “distinguishable only by subtle traits” but neither list the traits nor did they include individuals of Botrychium lunaria s.s. from Canada in their molecular study. Until recently (Stensvold & Farrar, 2016) it was unclear if Botrychium neolunaria and Botrychium lunaria s.s are genetically distinct in Canada, and their distributions are not known with precision. Botrychium neolunaria and Botrychium lunaria var. lunaria from multiple non-Canadian jurisdictions were shown to be distinct genetically by Dauphin, Vieu & Grant (2014); they also identified multiple geographically-grouped clusters of individuals of Botrychium lunaria var. lunaria. A psbA-trnH sequence of one of our collections (3755b) is identical to those of plants identified as Botrychium neolunaria from Alaska and Washington (Fig. 15). Four of our rbcL sequences are part of the Lunaria cluster, but there is insufficient variation for the region to be useful in species identification within this lineage. Two of our new rbcL sequences (Gillespie 9209, 9281) are identical to all published rbcL sequences of Botrychium lunaria s.l., and one of our samples (4179b) differs from these sequences by one substitution (Fig. 15). We identify these three samples as Botrychium tunux (see below). The sample (3755b) that clusters with Botrychium neolunaria in the psbA-trnH tree is similarly identical to all but one of the other accessions of the Botrychium lunaria complex in the rbcL tree. The samples of Botrychium lunaria s.l. in the tree likely correspond to Botrychium neolunaria, Botrychium tunux, and possibly Botrychium lunaria s.s., but clarification of this awaits careful study of the voucher specimens.

Re-examination of collection 3755b with the molecular placements in mind confirms its identity as a member of the Botrychium lunaria complex, and given its placement in the psbA-trnH tree we identify it as Botrychium neolunaria. This is the first confirmed record of Botrychium neolunaria for Nunavut. This single plant was growing among a population of Botrychium tunux (3755a), of which numerous individuals were gathered. We did not consider there to be two taxa growing together when sampling the site, and although we collected only one individual of Botrychium neolunaria, it is possible there were more present. Morphological characters that differentiate these species are given in the key below, with the caveat that the characters for Botrychium neolunaria may also correspond to Botrychium lunaria s.s. in Canada. These moonworts grew on a densely vegetated sik-sik den (a disturbed habitat) with Arnica angustifolia, Carex podocarpa, Dasiphora fruticosa, Elymus alaskanus, Oxyria digyna, Poa alpina, Salix reticulata, Stellaria longipes, Symphyotrichum pygmaeum, and Trisetum spicatum. A collection from the Big Bend area of the Coppermine River (Reading 1, DAO-784461 01-01000677497) was also re-determined to Botrychium neolunaria. According to the revised taxonomy of Botrychium lunaria s.l., many other collections of the complex from North America are likely referable to Botrychium neolunaria. Several Arctic collections of Botrychium lunaria are reported for northern Quebec and Labrador (Dignard, 2013e) and two collections have been recorded from the forest tundra in southeastern mainland Nunavut (Cody, 1996a).

Specimens Examined: Canada. Nunavut: Kitikmeot Region: flats atop and upper slopes of Coppermine Mountains, N/W side of Coppermine River, 67°14′53.6″N, 115°38′37.9″W ± 15 m, 401 m, 9 July 2014, Saarela, Sokoloff & Bull 3755b (CAN).

Botrychium tunux Stensvold & Farrar, Fig. 16—Tunux moonwort | Noteworthy Record

Figure 16 Botrychium tunux.

(A) Sporophore, Saarela et al. 3755a. (B) Sporophore, Saarela et al. 4179b. (C) Habitat, Coppermine Mountains, northwest of Coppermine River, Nunavut, 9 July 2014. Photographs by J. M. Saarela (A, C) and P. C. Sokoloff (B).

This species was segregated from Botrychium lunaria s.l. on the basis of genetic and morphological evidence, and described from the Yakatut area of Alaska (Stensvold, Farrar & Johnson-Groh, 2002), at the time the only known area for the species. It has since been recorded from Yukon (Stensvold & Farrar, 2008), British Columbia (E-Flora, 2015), Montana (Mincemoyer, 2013), Idaho, Colorado (Snow, 2009), New Mexico (Legler, 2010), Nevada and California (Rundel, 2011; Farrar, 2011). Dauphin, Vieu & Grant (2014) found that individuals of Botrychium tunux formed a distinct subclade within a broader Lunaria clade. The three individuals they sampled were from Yukon, Colorado and Alaska. In the psbA-trnH tree, our collections from Northwest Territories (Gillespie et al. 9281, 9209) and one of our Coppermine collections (4179b) group with these three Botrychium tunux accessions (Fig. 16). There is insufficient variation in rbcL for the region to be useful in species identification within the Lunaria lineage, as summarised under Botrychium neolunaria. Morphological characteristics distinguishing Botrychium tunux from Botrychium minganense and Botrychium neolunaria are given in the key below, with the caveat that the characters for Botrychium neolunaria may also correspond to Botrychium lunaria s.s. in Canada. Re-examination of our Coppermine and Northwest Territories plants with the psbA-trnH placements in mind confirmed their identification as Botrychium tunux. The diagnostic characters for Botrychium tunux are evident in the image of Gillespie et al. 9281 in Saarela et al. (2013a: 34).

We gathered the species at two sites. Atop the Coppermine Mountains it grew with Botrychium neolunaria (see comments under that species), and on a slope on the east side of Bloody Falls it grew with Botrychium minganese (see comments under that species). Based on a photo taken by J. K. Hebden in 1980 (DAO), Cody (1996a) reported Botrychium lunaria from Bloody Falls, along the Portage Trail (then on the east side of the river). Based on the image, we identify the species as Botrychium tunux. The old Portage Trail is close to the site of our collection of Botrychium tunux, and it is possible the 1980 photo was taken in the same general area. An unpublished collection from north of Dismal Lakes and west of the Coppermine River (Reading 64, DAO) is mixed, comprising plants of Botrychium minganense and Botrychium tunux. These earlier collections and our new ones represent the first reports of Botrychium tunux for Northwest Territories and Nunavut, substantially extending the range of the species from the nearest known localities in southwestern Yukon. Clarification of the full northern distribution of Botrychium tunux awaits revision of herbarium material determined as Botrychium lunaria.

Specimens Examined: Canada. Nunavut: Kitikmeot region: Bloody Falls, Coppermine River, along Portage Trail, 67°44′N, 115°23′W, 2 August 1980, H. J. Hebden s.n. [photo] (DAO-664033 01-01000677961); flats atop and upper slopes of Coppermine Mountains, N/W side of Coppermine River, 67°14′53.6″N, 115°38′37.9″W ± 15 m, 401 m, 9 July 2014, Saarela, Sokoloff & Bull 3755a (CAN); shallow gully in sand hills above Bloody Falls, SE side of Coppermine River on SW-facing slopes, 67°44′28.2″N, 115°22′3″W ± 15 m, 78 m, 19 July 2014, Saarela, Sokoloff & Bull 4179b (CAN).

Key to Botrychium species in the Coppermine River Valley, adapted from Stensvold, Farrar & Johnson-Groh (2002) and Farrar (2011): 1. Basal pinnae narrowly fan-shaped, usually spanning an arc less than 120°; pinnae remoteBotrychium minganense

1′ Basal pinnae broadly fan-shaped, usually spanning an arc of 150–180°; pinnae approximate to overlapping2

2. Pinnae asymmetrical with the basal ones larger; sporophore stalk at spore release shorter than or equal to the length of the trophophore; plants 6–12 cm tallBotrychium tunux

2′ Pinnae symmetrical, sporophore stalk at spore release longer than trophophore; plants 8–25 cm tallBotrychium neolunaria

Pteridaceae [1/1]

Cryptogramma stelleri (S. G. Gmel.) Prantl., Fig. 17A—Steller’s rockbrake | European (NE)–Asian (N/C)–amphi-Beringian–Cordilleran & North American (NE) | Noteworthy Record

Figure 17 Cryptogramma stelleri and Woodsia glabella.

Cryptogramma stelleri: (A) habit, Saarela et al. 4076. Woodsia glabella: (B) habit, Saarela et al. 3891. Photographs by P. C. Sokoloff.

Newly recorded for the study area and mainland Nunavut, and the second record for Nunavut. First reported for Nunavut by Gillespie et al. (2015), based on a collection from Kimmirut on southern Baffin Island. We found one small patch of the rare species growing in a crack on a south-facing cliff just west of the start of Bloody Falls in Kugluk (Bloody Falls) Territorial Park. This represents a northeastern range extension of some 530 km from the nearest known locality from the Richardson Mountains west of Great Bear Lake (Cody & Porsild, 1968; Porsild & Cody, 1980; Cody & Britton, 1989). Further details on its scattered distribution are given in Gillespie et al. (2015).

Specimen Examined: Canada. Nunavut: Kitikmeot Region: Kugluk (Bloody Falls) Territorial Park, S-facing cliff (gabbro sill) above start of Bloody Falls, W side of Coppermine River, W side of Portage Trail, 67°44′23.2″N, 115°22′54.5″W ± 50 m, 57 m, 16 July 2014, Saarela, Sokoloff & Bull 4076 (CAN).

Woodsiaceae [1/3]

Woodsia alpina (Bolton) S. F. Gray—Alpine woodsia, alpine cliff brake | Circumpolar–alpine

Previously collected in Kugluktuk, on “dry ledges”. The collecting site was probably North Hill, as this is the only area near the community characterised by distinctive dry ledges. We also made a collection there, where we found it growing in cracks on northwest-facing rocky ledges. The earlier collection from Kugluktuk was mapped in Porsild & Cody (1980), but not in Cody & Britton (1989). Our second collection, from Bloody Falls (but outside Kugluk (Bloody Falls) Territorial Park), is a single plant that is mostly glabrous. This character state is more typical of Woodsia glabella, but the collection is placed here on the basis of having a few lanceolate scales along the mid-point of the rachis of one of the fronds. Woodsia glabella sometimes has a few scales proximally on the rachis, but not at mid-point. In the Canadian Arctic Archipelago recorded from southern Baffin Island, Coats, Ellesmere, Nottingham and Southampton islands (Aiken et al., 2007), and it has a scattered distribution across the mainland Arctic (Porsild & Cody, 1980; Korol, 1992; Cody, 1996a; Cody, Reading & Line, 2003; Dignard, 2013g). The dot map in Porsild & Cody (1980) appears to be the basis for the shaded range map in Alverson (1993), which covers most of mainland Nunavut.

Specimens Examined: Canada. Nunavut: Kitikmeot region: Coppermine [Kugluktuk], vicinity of post [67°49′36″N, 115°5′36″W ± 1.5 km], 26 July 1949, A. E. Porsild 17158 (CAN-127419); Kugluktuk, rocky slopes of North Hill, 67°49′31.4″N, 115°6′54″W ± 100 m, 42 m, 29 June 2014, Saarela, Sokoloff & Bull 3067 (CAN, UBC); SSW-facing slopes above start of Bloody Falls, SE side of Coppermine River, 67°44′12.5″N, 115°22′31″W ± 50 m, 50–60 m, 19 July 2014, Saarela, Sokoloff & Bull 4204 (CAN).

Woodsia glabella R. Br., Fig. 17B—Smooth cliff fern, smooth cliff brake | Circumpolar–alpine

Previously recorded from Kugluktuk (Porsild & Cody, 1980), although this earlier record was not mapped in Cody & Britton (1989). This is a common species of rocky slopes and cliffs along the Coppermine River valley. We made collections at Fockler Creek and Kugluk (Bloody Falls) Territorial Park. Widespread throughout the Canadian Arctic (Porsild & Cody, 1980; Korol, 1992; Cody, Reading & Line, 2003; Aiken et al., 2007; Dignard, 2013g; Saarela et al., 2013a).

Specimens Examined: Canada. Nunavut: Kitikmeot region: Coppermine [Kugluktuk], vicinity of post [67°49′36″N, 115°5′36″W ± 1.5 km], 26 July 1949, A. E. Porsild 17159 (CAN-127423); N side of Fockler Creek, ca. 1.9 km S of Sandstone Rapids, Coppermine River, 67°25′57.89″N, 115°38′3.9″W ± 10 m, 162 m, 4 July 2014, Saarela, Sokoloff & Bull 3319 (CAN); esker on E side of Coppermine River, 0.6 km SSE of Muskox Rapids, 67°22′40″N, 115°42′38.5″W ± 50 m, 172 m, 8 July 2014, Saarela, Sokoloff & Bull 3664 (CAN); flats atop and upper slopes of Coppermine Mountains, N/W side of Coppermine River, 67°14′49.9″N, 115°38′43.7″W ± 200 m, 467 m, 9 9 July 2014, Saarela, Sokoloff & Bull 3773 (CAN); Kugluk (Bloody Falls) Territorial Park, rocky valley immediately SW of Bloody Falls, along rough marked section of Portage Trail, 67°44′34″N, 115°22′16″W ± 50 m, 20 m, 13 July 2014, Saarela, Sokoloff & Bull 3891 (CAN); Kugluk (Bloody Falls) Territorial Park, S-facing cliff (gabbro sill) above start of Bloody Falls, W side of Coppermine River, W side of Portage Trail, 67°44′23.2″N, 115°22′54.5″W ± 50 m, 57 m, 16 July 2014, Saarela, Sokoloff & Bull 4082 (CAN); Kugluk (Bloody Falls) Territorial Park, rocky valley immediately SW of Bloody Falls, along rough marked section of Portage Trail, 67°44′34″N, 115°22′16″W ± 50 m, 20 m, 18 July 2014, Saarela, Sokoloff & Bull 4158 (CAN, UBC).

Woodsia ilvensis (L.) R. Br.—Rusty woodsia | Circumboreal-polar

Mapped for the study area in Porsild & Cody (1980), but not in Cody & Britton (1989). There is a single collection from Kugluktuk, but a precise location is not recorded on the specimen label. We did not collect it in 2014. Elsewhere in the Canadian Arctic recorded from southern Baffin Island and Nottingham Island, a few other sites on mainland Nunavut, and northern Quebec and Labrador (Porsild & Cody, 1980; Cody, Scotter & Zoltai, 1984; Cody, Reading & Line, 2003; Cody & Reading, 2005; Aiken et al., 2007; Dignard, 2013g).

Specimens Examined: Canada. Nunavut: Coppermine [Kugluktuk] [67.826667°N, 115.09333°W ± 1.5 km], 5 July 1958, R. D. Wood s.n. (CAN-265609).

Gymnosperms

Cupressaceae [1/1]

Juniperus communis subsp. depressa (Pursh) Franco, Figs. 18A and 18B—Common juniper | North American (N) | Noteworthy Record

Figure 18 Juniperus communis subsp. depressa and Picea glauca.

Juniperus communis subsp. depressa: (A) habitat, Saarela et al. 3642. (B) Cones, vicinity of lower Coppermine River, Nunavut, 7 July 2014. Picea glauca: (C) cones, vicinity of Fockler Creek, Nunavut, 4 July 2014. (D) Habitat, vicinity of Fockler Creek, Nunavut, 5 July 2014. Photographs by J. M. Saarela (A) and R. D. Bull (B, C, D).

Newly recorded for the study area. We made collections at Fockler Creek, Melville Creek, Big Creek, Coppermine Mountains, and Kugluk (Bloody Falls) Territorial Park. Previously reported from the nearby Big Bend area of the Coppermine River (Reading 32, DAO; Cody, Reading & Line, 2003) and a site ca. 15 km east of the confluence of the Kendall and Coppermine rivers (Reading 83, DAO; Cody & Reading, 2005). Our collections represent a minor northern range extension with respect to these. This prostrate shrub is common along the banks of the Coppermine River, where it is often found with white spruce, and in Kugluk (Bloody Falls) Territorial Park, where it grows along the bottom and lower wet edges of a creek valley. It may occur further north along the Coppermine River, but was not found at Kugluktuk. It has been recorded north of the treeline at a few sites in Northwest Territories: Tuktut Nogait National Park and vicinity (Saarela et al., 2013a) and the southeastern portion of the Thelon Game Sanctuary (several collections at CAN). All of these Arctic collections are outside the range in recent distribution maps (Adams, 1993, 2008; Farjon & Filer, 2013). It is also known from southeastern mainland Nunavut (Porsild, 1950b; Porsild & Cody, 1980). It is not recorded from the Canadian Arctic Archipelago. Subspecies depressa is the common, widespread North American subspecies (Elven et al., 2011). Arctic populations of Juniperus communis from southern Greenland were shown to have colonised that island via long-distance dispersal from Europe, while populations of the species from across northern Canada were genetically similar (Alsos et al., 2015).

Specimens Examined: Canada. Nunavut: Kitikmeot region: old riverbed of Fockler Creek, ca. 2.3 km SSE of Sandstone Rapids, Coppermine River, 67°25′48″N, 115°37′33″W ± 25 m, 153 m, 1 July 2014, Saarela, Sokoloff & Bull 3141 (CAN, QFA, US, WIN); confluence of Coppermine River and Melville Creek, just W of Coppermine Mountains, 67°15′52″N, 115°30′55.3″W ± 350 m, 178–190 m, 7 July 2014, Saarela, Sokoloff & Bull 3526 (CAN, MO, MT); forest and slopes at confluence of Big Creek and Coppermine River, N side of Coppermine River, S side of Coppermine Mountains, 67°14′29.3″N, 116°2′44.5″W ± 250 m, 180–199 m, 7 July 2014, Saarela, Sokoloff & Bull 3568 (CAN, UBC); S-facing slopes above Coppermine River, ca. 7.8 km NNE of Sandstone Rapids, 67°31′16.2″N, 115°36′52.1″W ± 50 m, 110 m, 8 July 2014, Saarela, Sokoloff & Bull 3642 (ALA, CAN); flats atop and upper slopes of Coppermine Mountains, N/W side of Coppermine River, 67°36′58.7″N, 115°29′18.3″W ± 99 m, 50 m, 8 July 2014, Saarela, Sokoloff & Bull 3782 (ALTA, CAN); Kugluk (Bloody Falls) Territorial Park, gentle stream in shallow valley running into Coppermine River just W of Bloody Falls, 67°44′36.6″N, 115°22′59.3″W ± 20 m, 41 m, 15 July 2014, Saarela, Sokoloff & Bull 4015 (CAN, O).

Pinaceae [1/1]

Picea glauca (Moench) Voss, Figs. 18C and 18D—White spruce | North American (N)

The northern spruce groves along the lower Coppermine River have been of interest since first seen by western explorers. They were noted in accounts by Hearne (1795), Franklin (1823) and Richardson (1851). The first collection of white spruce from the study area was taken from Sandstone Rapids (Macoun & Holm, 1921), as Picea canadensis (Mill.) B. S. P., in February 1915 as part of the Canadian Arctic Expedition 1913–1918. There is a handwritten note on a copy of the report in the reprint library at CAN that the collection number is 720. We were unable to find any relevant specimens at CAN. Spruce development at Escape Rapids and Sandstone Rapids was described by Johansen (1919). He also commented on spruce trees along the Napaaktoktok River, about 10 km east of the study area, based on information presented to him by R. M. Anderson: spruce trees “in the small, more unprotected Naparkoktuak [Napaaktoktok] River (its name is Eskimo [sic] for spruce), is very stunted (below six feet) and scattered, only a little grove of trees being found here and there; but the trees reach to within a dozen miles of the Arctic coast” (Johansen, 1919: 304) (also see Anderson, 1917). Further details on spruce in the area are given by (Holm, 1922: 85B). Stefansson (1912) noted “the most northerly sprigs of growing spruce are a mile and a half north of Bloody Fall, or less than six miles from the sea in an airline”. We are not aware of any other reports of spruce growing this far north along the Coppermine River. Spruce stands at Escape Rapids and further south are described by Johansen (1924: 49c); also see Johansen (1919).

We made collections at Fockler Creek, Melville Creek, Escape Rapids, and Kugluk (Bloody Falls) Territorial Park, where we found one spruce tree growing in a dense willow thicket on the west side of the Coppermine River near the northern park boundary. In the large spruce forest at Fockler Creek, many trees had been cut several feet from their base (likely in winter). Few authoritative distribution maps of white spruce record the species along the lower Coppermine River (Ritchie, 1987; Farrar, 1995). Elsewhere on mainland Arctic Nunavut recorded from just north of the Manitoba border along Hudson Bay (Porsild & Cody, 1980). In northern Quebec white spruce barely extends into the Arctic ecozone along the Hudson Bay coast near Umiujaq and there are two records for northern Labrador (Caccianiga & Payette, 2006; Dignard, 2013f). It is not recorded from the Canadian Arctic Archipelago (Aiken et al., 2007).

Specimens Examined: Canada. Nunavut: Kitikmeot region: S of Fockler Creek, S-facing slope on N side of small tributary of Fockler Creek, ca. 2.3 km S of Sandstone Rapids, Coppermine River, 67°25′46.3″N, 115°38′2.5″W ± 5 m, 156 m, 6 July 2014, Saarela, Sokoloff & Bull 3458 (ALA, CAN, UBC); confluence of Coppermine River and Melville Creek, just W of Coppermine Mountains, 67°15′52″N, 115°30′55.3″W ± 350 m, 178–190 m, 7 July 2014, Saarela, Sokoloff & Bull 3520 (CAN); S-facing slopes above Coppermine River, ca. 7.8 km NNE of Sandstone Rapids, 67°31′16.2″N, 115°36′52.1″W ± 50 m, 110 m, 8 July 2014, Saarela, Sokoloff & Bull 3639 (CAN); SE-facing slopes above Escape Rapids, W side of Coppermine River, 67°36′58.7″N, 115°29′18.3″W ± 99 m, 50 m, 8 July 2014, Saarela, Sokoloff & Bull 3730 (CAN); Kugluk (Bloody Falls) Territorial Park, W side of Coppermine River, just above Bloody Falls, 67°44′22.6″N, 115°22′52″W ± 20 m, 40 m, 16 July 2014, Saarela, Sokoloff & Bull 4105 (CAN).

Monocots

Amaryllidaceae [1/1]

Allium schoenoprasum L. “northern race”, Fig. 19—Wild chives | European (N)–Asian (N)–amphi-Beringian–North American (N) | Noteworthy Record

Figure 19 Allium schoenoprasum.

(A) Habit, Saarela et al. 3935. (B) Inflorescence, Saarela et al. 3935. (C) Habitat, Saarela et al. 3935. Photographs by R. D. Bull (A) and P. C. Sokoloff (B, C).

Wild chives is a wide ranging, morphologically variable circumpolar taxon, and the only one of the approximately 800 known onion species that definitely reaches the Arctic (one other species, Allium strictum Schrad., reaches the borderline Arctic in Siberia) (Elven et al., 2011). We made three collections, representing a considerable range extension. The largest population encountered was in Kugluk (Bloody Falls) Territorial Park, growing on the upper ledges of the cliffs above the start of Bloody Falls rapids with Anthoxanthum monticola, Anthoxanthum angustifolia, Calamagrostis purpurascens, Dryopteris fragrans, Poa glauca, and Saxifraga tricuspidata. This population is just steps away from the adjacent Portage Trail used by hikers and paddlers and it is surprising this conspicuous species has not previously been collected or noted at the site. Another collection was gathered in a very different Arctic habitat: grassy sandy flats on an extensive sandy floodplain of the Coppermine River near Kugluktuk, growing with Astragalus alpinus, Bromus pumpellianus, Castilleja caudata, Eurybia sibirica, Parnassia palustris, Salix niphoclada, Salix alaxensis, and Symphyotrichum pygmaeum. Our third collection was made in a Subarctic spruce forest community.

The nearest records to the southwest are from southwestern Great Bear Lake, and to the south from just north of Great Slave Lake (Porsild & Cody, 1980). McNeal & Jacobsen (2002) recorded the species for Nunavut, but the source of this report is unclear to us. There are no collections mapped for Nunavut in Porsild & Cody (1980) or reported, to our knowledge, in the literature. Pending confirmation of the report in McNeal & Jacobsen (2002), we consider our collections to be the first confirmed records for Nunavut. This is the only species of the family Amaryllidaceae for the territory. Elsewhere in the Canadian Arctic known from Yukon on Herschel Island (Kennedy s.n., Environmental Yukon Herbarium, photo DAO) and the adjacent mainland along the Firth River (Cody et al., 2004) and Ivvavik National Park (Sheep Creek drainage, 69.135°N, 140.15°W, 270 m, Bennett et al. 08-175, CAN-590221), and Northwest Territories (Anderson River Delta, 69°42′N, 129°W, Barry 156, CAN-287908). To the southwest, the nearest collections are from the Great Bear River area (Mount Charles, 65°04′N, 124°37′W, 16 June 1928, Porsild & Porsild 3309, CAN-13382; junction with Big Stick River, 64°57′N, 123°48′W, 15–16 June 1928, Porsild & Porsild 3253, CAN-13381). The taxonomy of the species is controversial. Several authors in North America have referred native plants (wild chives) to Allium schoenoprasum var. sibiricum (L.) Hartm. (Hultén, 1968; Porsild & Cody, 1980; Kwiatkowski, 1999; Cody, 2000; Reznicek, Voss & Walters, 2011), and sometimes distinguishing them from introduced plants (garden chives) treated as var. schoenoprasum (e.g., Reznicek, Voss & Walters, 2011). However, the name “sibiricum” and its application are problematic. As noted by Friesen (1996) and Elven et al. (2011), Linnaeus (1771) described Allium sibiricum as a white-flowered species different from the taxon that subsequent workers have referred to by the same name with pink to purple flowers. They concluded it is incorrect to apply the name in this misapplied sense. Elven et al. (2011) recognised the distinctiveness of Arctic-Subarctic plants of Allium schoenoprasum s.l. (as “northern race”) on the basis of morphological and ecological characters, stated that these northern plants require a valid name if recognised, and did not consider Allium sibiricum and other combinations as synonyms of Allium schoenoprasum.

Other authors in North America and elsewhere do not recognise infraspecific taxa in Allium schoenoprasum and include in its synonymy Allium sibiricum and homotypic synonyms (McNeal & Jacobsen, 2002; Czerepanov, 2007; Choi & Cota-Sánchez, 2010; Haines et al., 2011). Friesen (1996), in his global revision of Allium sect. Schoenoprasum Dumort., treated Allium sibiricum as a synonym of Allium schoenoprasum subsp. schoenoprasum, one of three subspecies he recognised and the only one recognised in North America. Friesen (1996) noted four informal morphotypes in this taxon, one of which (“Type C”) has usually been referred to as Allium sibiricum. It is not clear if Friesen’s Type C corresponds to the “northern race” of Elven et al. (2011). If so, it is possible one of the valid names listed under his Type C may apply to the Arctic-Subarctic race. Whatever the correct name, our plants correspond to Elven et al.’s (2011) northern race in their leaf blades (within the 2–6 mm wide range and extending up the stem, at least in Saarela et al. 3935), but tepal length (7–10 mm) falls within the range reported for Allium schoenoprasum s.s., not the northern race (ca. 15 mm); capsules are not available on our material to evaluate variation in that character as noted by Elven et al. (2011).

The Linnaean Plant Name Typification Project website indicates “type not designated” for Allium sibiricum and lists Herb. Linn. No. 419.38, a white-flowered individual, as possible original material. Choi & Cota-Sánchez (2010) noted LINN 419.38a to be the lectotype of Allium sibiricum; however, they neither indicate who designated this lectotype nor does their statement meet the requirements of Article 7.10 for designating a lectotype (i.e., “designated here” not specified).

Specimens Examined: Canada. Nunavut: Kitikmeot region: forest and slopes at confluence of Big Creek and Coppermine River, N side of Coppermine River, S side of Coppermine Mountains, 67°14′29.3″N, 116°2′44.5″W ± 250 m, 180–199 m, 7 July 2014, Saarela, Sokoloff & Bull 3560 (CAN); Kugluk (Bloody Falls) Territorial Park, upper ledges of rocky (gabbro) S-facing cliffs above the start of Bloody Falls (W bank of River), just E of Portage Trail, 67°44′21.7″N, 115°22′42.2″W ± 25 m, 46 m, 14 July 2014, Saarela, Sokoloff & Bull 3935 (ALA, CAN, UBC); grassy sandy flats on extensive sandy floodplain of Coppermine River, below steep cliff above river and S of Kugluktuk, 67°48′54.3″N, 115°6′9.1″W ± 20 m, 5 m, 26 July 2014, Saarela, Sokoloff & Bull 4416 (CAN).

Cyperaceae [4/47]

Carex adelostoma V. I. Krecz.—Circumpolar sedge | Circumboreal-polar? | Noteworthy Record

Our two collections of this rhizomatous sedge are the first for the study area and represent a northern range extension. They are the first collections from Nunavut in 84 years. Near Fockler Creek, this species was locally common in a large sedge wetland growing with Carex microglochin and Eriophorum angustifolium. At a site just north of the southern limit of the Arctic ecozone, it was common in a wet area with Betula glandulosa, Saussurea angustifolia and Salix spp. Elsewhere in Nunavut known only from Yathkyed Lake on the Kazan River, where Porsild collected it in 1930 (Porsild, 1943; Porsild & Cody, 1980). In Northwest Territories known from several collections from northern and eastern Great Bear Lake (Porsild, 1943). In Alaska known from four to five localities in the Tanana-Kuskokwim Lowlands, Copper River Basin, and Wrangell Mountains (Cook & Roland, 2002) where it is considered a rare plant (Nawrocki, Fulkerson & Carlson, 2013). Porsild & Cody (1980) recognised this taxon as Carex morrisseyi Porsild (Porsild, 1943), a name treated as a synonym of Carex adelostoma by Murray (2002b) and as a tentative synonym by Elven et al. (2011). It is considered rare in Northwest Territories (Cody, 1979), Quebec (Bouchard et al., 1983) and Canada (Argus & Pryer, 1990).

Specimens Examined: Canada. Nunavut: Kitikmeot region: flats N of Fockler Creek, between Fockler Creek and Sleigh Creek, ca. 0.6 km SE of Sandstone Rapids, Coppermine River, 67°26′26″N, 115°37′40.7″W ± 3 m, 125 m, 6 July 2014, Saarela, Sokoloff & Bull 3486 (ALTA, CAN, MO, MT, O); wet area on flats above W bank of Coppermine River, ca. 7.9 km NNE of Sandstone Rapids, 67°31′18.3″N, 115°36′48.9″W ± 5 m, 117 m, 8 July 2014, Saarela, Sokoloff & Bull 3654 (ALA, CAN, NY, UBC).

Carex aquatilis subsp. stans (Drejer) Hultén, Figs. 20A and 20B—Aquatic sedge | Circumpolar-alpine

Figure 20 Carex aquatilis subsp. stans and C. bicolor.

Carex aquatilis subsp. stans: (A) habit, Saarela et al. 3585. (B) Habitat, Saarela et al. 3585. Carex bicolor: (C) inflorescences, Saarela et al. 3857. (D) Habit, Saarela et al. 3857. Photographs by J. M. Saarela.

Previously recorded from Kugluktuk (Cody, 1954b; Porsild & Cody, 1980), as Carex aquatilis Wahlenb. We made collections at Fockler Creek, Melville Creek, Kendall River, Kugluk (Bloody Falls) Territorial Park and Kugluktuk. The taxonomy of the Carex aquatilis complex in the Arctic is reviewed in Saarela et al. (2013a); subsp. stans (syn. Carex aquatilis var. minor Boott) is the Arctic race of the species. Widespread throughout the Canadian Arctic (Porsild & Cody, 1980; Cody, Scotter & Zoltai, 1989; Korol, 1992; Aiken et al., 2007; Saarela et al., 2013a).

Specimens Examined: Canada. Nunavut: Kitikmeot Region: Coppermine [Kugluktuk], 67°49′36″N, 115°5′36″W, 3 July 1951, W. I. Findlay 74 (MT00183522); sedge meadow at S end of small lake, on flats NW of Fockler Creek, ca. 1.9 km SSE of Sandstone Rapids, Coppermine River, 67°26′1.8″N, 115°37′30.5″W ± 20 m, 170 m, 2 July 2014, Saarela, Sokoloff & Bull 3236 (CAN, K, QFA, US, WIN); confluence of Coppermine River and Melville Creek, just W of Coppermine Mountains, 67°15′52″N, 115°30′55.3″W ± 350 m, 178–190 m, 7 July 2014, Saarela, Sokoloff & Bull 3496 (CAN, UBC); confluence of Coppermine and Kendall rivers (NW side of Coppermine River, S side of Kendall River), small ponds on flats adjacent to Coppermine River, 67°6′44.7″N, 116°8′6.1″W ± 100 m, 213 m, 7 July 2014, Saarela, Sokoloff & Bull 3585 (ALTA, CAN, O); Kugluk (Bloody Falls) Territorial Park, rocky valley immediately SW of Bloody Falls, along rough marked section of Portage Trail, head of small unnamed pond just W of falls, 67°44′42.8″N, 115°22′29.2″W ± 10 m, 9 m, 13 July 2014, Saarela, Sokoloff & Bull 3900 (CAN, MO, MT); W of Kugluktuk on tundra flats above Coppermine River, S of 1 Coronation Drive and N of power plant, 67°49′28.97″N, 115°5′0.2″W ± 100 m, 8 m, 25 July 2014, Saarela, Sokoloff & Bull 4368 (ALA, CAN).

Carex atrofusca Schkuhr—Dark brown sedge | Circumpolar–alpine

Previously recorded from Kugluktuk (Porsild & Cody, 1980). We made collections at Fockler Creek, Melville Creek and Kugluk (Bloody Falls) Territorial Park. Widespread throughout the Canadian Arctic (Porsild & Cody, 1980; Korol, 1992; Aiken et al., 2007; Saarela et al., 2013a).

Specimens Examined: Canada. Nunavut: Kitikmeot Region: Coppermine [Kugluktuk], vicinity of post [67.826667°N, 115.09333°W ± 1.5 km], 26 July 1949, A. E. Porsild 17161 (CAN-127536); Coppermine [Kugluktuk] [67°49′36″N, 115°5′36″W ± 1.5 km], 2 July 1958, R. D. Wood s.n. (CAN-265601); sedge meadow adjacent to small lake on flats N of Fockler Creek, ca. 1.5 km SSE of Sandstone Rapids, Coppermine River, 67°26′8.8″N, 115°37′35.9″W ± 20 m, 168 m, 2 July 2014, Saarela, Sokoloff & Bull 3223 (CAN, UBC); confluence of Coppermine River and Melville Creek, just W of Coppermine Mountains, 67°15′52″N, 115°30′55.3″W ± 350 m, 178–190 m, 7 July 2014, Saarela, Sokoloff & Bull 3495 (CAN); Kugluk (Bloody Falls) Territorial Park, small rocky meadow along small stream that runs into Coppermine River just below Bloody Falls, about 1 km W of Bloody Falls, 67°44′40.1″N, 115°23′37.5″W ± 15 m, 48 m, 15 July 2014, Saarela, Sokoloff & Bull 4027 (ALA, CAN, WIN); NW-facing moist to wet sedge meadow, drainage running into Coppermine River above Bloody Falls on SE side, 67°44′26.2″N, 115°22′11.8″W ± 15 m, 47 m, 19 July 2014, Saarela, Sokoloff & Bull 4192 (ALTA, CAN, MO, O).

Carex bicolor Bellardi ex All., Figs. 20C and 20D—Bicoloured sedge | Circumpolar–alpine | Noteworthy Record

Newly recorded for the study area. We made collections of this densely cespitose sedge at Fockler Creek, Heart Lake, Kugluk (Bloody Falls) Territorial Park and Kugluktuk. These close a distribution gap between sites at Bathurst Inlet, Hood River, northeastern Great Bear Lake and Tuktut Nogait National Park and vicinity (Porsild, 1943; Porsild & Cody, 1980; Gould & Walker, 1997; Saarela et al., 2013a; Bennett, 2015). Elsewhere in the Canadian Arctic recorded from southern Baffin Island, Coats, Nottingham and Victoria islands, and a few sites on mainland Nunavut and Northwest Territories (Porsild & Cody, 1980; Cody, Scotter & Zoltai, 1989; Aiken et al., 2007; Saarela et al., 2013a; Gillespie et al., 2015).

Specimens Examined: Canada. Nunavut: Kitikmeot Region: S of Fockler Creek, along small tributary that runs into Fockler Creek, ca. 2.3 km S of Sandstone Rapids, Coppermine River, 67°25′44.9″N, 115°38′25.9″W ± 100 m, 152 m, 3 July 2014, Saarela, Sokoloff & Bull 3263 (CAN, MO, MT); S of Fockler Creek, along small tributary that runs into Fockler Creek, ca. 2.3 km S of Sandstone Rapids, Coppermine River, 67°25′44.9″N, 115°38′25.9″W ± 100 m, 152 m, 3 July 2014, Saarela, Sokoloff & Bull 3277 (CAN, UBC, US); small unnamed lake N of Fockler Creek, between Fockler and Sleigh Creek, ca. 1.4 SSE of Sandstone Rapids, Coppermine River, 67°26′13.4″N, 115°37′48″W ± 5 m, 165 m, 6 July 2014, Saarela, Sokoloff & Bull 3487 (ALA, CAN, UBC, US, WIN); E end of small, unnamed lake on W bank of Coppermine River, ca. 8.3 km NNE of Sandstone Rapids, 67°31′30.8″N, 115°36′16.1″W ± 50 m, 126 m, 8 July 2014, Saarela, Sokoloff & Bull 3661 (ALA, CAN); Heart Lake, SW of Kugluktuk, 6.4 km SW of mouth of Coppermine River, 67°49′29.2″N, 115°1′3.2″W ± 50 m, 1 m, 8 July 2014, Saarela, Sokoloff & Bull 3726 (CAN, K, NY, QFA, UBC, WIN); SE-facing slopes above Escape Rapids, W side of Coppermine River, 67°36′49.8″N, 115°29′27.4″W ± 10 m, 67 m, 8 July 2014, Saarela, Sokoloff & Bull 3742 (ALTA, CAN); Kugluk (Bloody Falls) Territorial Park, rocky valley immediately SW of Bloody Falls, along rough marked section of Portage Trail, 67°44′34″N, 115°22′16″W ± 50 m, 20 m, 13 July 2014, Saarela, Sokoloff & Bull 3857 (CAN, QFA, WIN); Kugluk (Bloody Falls) Territorial Park, rocky beach above Bloody Falls, W bank of Coppermine River, 67°44′18″N, 115°22′57.3″W ± 250 m, 34 m, 14 July 2014, Saarela, Sokoloff & Bull 3978 (CAN, K, NY); W of Kugluktuk on tundra flats above Coppermine River, S of 1 Coronation Drive and N of community power plant, 67°49′28.97″N, 115°5′0.2″W ± 100 m, 8 m, 22 July 2014, Saarela, Sokoloff & Bull 4273 (CAN, O); grassy sandy flats on extensive sandy floodplain of Coppermine River, below steep cliff above river and S of Kugluktuk, 67°48′54.3″N, 115°6′9.1″W ± 20 m, 5 m, 26 July 2014, Saarela, Sokoloff & Bull 4417 (ALTA, CAN, MO, MT, O).

Carex bigelowii Torr. ex Schwein. subsp. bigelowii—Bigelow’s sedge | North American–amphi-Atlantic

Previously recorded from Kugluktuk (Cody, 1954b; Porsild & Cody, 1980), as Carex bigelowii s.s. Our two collections from Kugluk (Bloody Falls) Territorial Park are from the same population, but were collected as different numbers because of morphological differences observed in the field. This species typically has staminate terminal spikelets, as in no. 4096, but no. 4097 has gynecandrous terminal spikelets. The subspecies bigelowii differs from subsp. lugens, which is more common in the study area, by having spotted purple-black perigynium apices (vs. uniformly purple black on apical half). It reaches its known western limit in the study area and is recorded elsewhere in the Canadian Arctic from numerous mainland sites as well as Baffin, Devon, Ellesmere, Southampton and Victoria islands (Porsild & Cody, 1980; L. J. Gillespie & J. M. Saarela, 2016, unpublished data; Cody, Scotter & Zoltai, 1989; Korol, 1992; Aiken et al., 2007). The problematic taxonomy of the Carex bigelowii complex is reviewed in Saarela et al. (2013a). Taxonomy here follows Standley, Cayouette & Bruederle (2002).

Specimens Examined: Canada. Nunavut: Kitikmeot Region: Coppermine [Kugluktuk], 67°49′36″N, 115°5′36″W, W. I. Findlay 53 (DAO-175578); Kugluk (Bloody Falls) Territorial Park, along wet, muddy and deeply pitted ATV trail ca. 1 km W of Bloody Falls, 67°44′33.2″N, 115°23′30″W ± 20 m, 73 m, 16 July 2014, Saarela, Sokoloff & Bull 4096 (ALA, ALTA, CAN, O); Kugluk (Bloody Falls) Territorial Park, along wet, muddy and deeply pitted ATV trail ca. 1 km W of Bloody Falls, 67°44′33.2″N, 115°23′30″W ± 20 m, 73 m, 16 July 2014, Saarela, Sokoloff & Bull 4097 (CAN, UBC).

Carex bigelowii subsp. lugens (Holm) T. V. Egorova—Spruce muskeg sedge | Eurasian–amphi-Beringian

Previously recorded from Kugluktuk, as Carex lugens Holm (Porsild & Cody, 1980). We made collections at Fockler Creek, Kugluk (Bloody Falls) Territorial Park and Kugluktuk. This subspecies is much more common than subsp. bigelowii in the study area. Elsewhere in the Canadian Arctic recorded from Banks and Victoria islands and some other mainland sites, reaching its known eastern limit in central mainland Nunavut (Porsild & Cody, 1980; Aiken et al., 2007; Saarela et al., 2013a). Taxonomy here follows Standley, Cayouette & Bruederle (2002).

Specimens Examined: Canada. Nunavut: Kitikmeot Region: Coppermine [Kugluktuk], mouth of Coppermine River [67.819444°N, 115.06389°W ± 2,000 m], 2 August 1962, J. A. Larsen s.n. (CAN-286314); flats on W side of Fockler Creek, above spruce forest in creek valley, ca. 2.2 km S of Sandstone Rapids, Coppermine River, 67°25′49″N, 115°37′55″W ± 50 m, 152 m, 1 July 2014, Saarela, Sokoloff & Bull 3113 (ALA, ALTA, CAN, MO, MT, O); S of Fockler Creek, above small tributary of Fockler Creek, ca. 2.3 km S of Sandstone Rapids, Coppermine River, 67°25′46.3″N, 115°38′2.5″W ± 100 m, 156 m, 6 July 2014, Saarela, Sokoloff & Bull 3453 (CAN, QFA, US, WIN); Kugluk (Bloody Falls) Territorial Park, along Portage Trail at top of ridge on W bank of Coppermine River, near start of Bloody Falls rapids, 67°44′22.5″N, 115°22′40.6″W ± 10 m, 46 m, 14 July 2014, Saarela, Sokoloff & Bull 3928 (CAN, MT, UBC); Kugluk (Bloody Falls) Territorial Park, upper ledges of rocky (gabbro) S-facing cliffs above the start of Bloody Falls (W bank of River), just E of Portage Trail, 67°44′21.7″N, 115°22′42.2″W ± 25 m, 46 m, 14 July 2014, Saarela, Sokoloff & Bull 3948 (CAN); Kugluk (Bloody Falls) Territorial Park, flats above boardwalk W of Bloody Falls, 67°44′34.5″N, 115°22′27″W ± 100 m, 135 m, 16 July 2014, Saarela, Sokoloff & Bull 4062 (CAN, K, NY, UBC); Kugluk (Bloody Falls) Territorial Park, W side of Coppermine River, between Sandy Hills and Bloody Falls, 67°45′13.2″N, 115°22′6.3″W ± 3 m, 21 m, 17 July 2014, Saarela, Sokoloff & Bull 4146 (CAN, MO); W of Kugluktuk on tundra flats above Coppermine River, S of 1 Coronation Drive and N of community power plant, 67°49′28.97″N, 115°5′0.2″W ± 100 m, 8 m, 22 July 2014, Saarela, Sokoloff & Bull 4275 (CAN).

Carex borealipolaris S. R. Zhang—Siberian kobresia | Asian (N/C)–amphi-Beringian

Previously recorded from Kugluktuk (Porsild & Cody, 1980). We made collections at Fockler Creek, Coppermine Mountains, Kugluk (Bloody Falls) Territorial Park and Kugluktuk. Elsewhere in the Canadian Arctic recorded from Baffin (one record) and Victoria islands, a few other mainland Nunavut sites and mainland Northwest Territories (Porsild & Cody, 1980; Aiken et al., 2007; Saarela et al., 2013a). This species was formerly recognised as Kobresia sibirica (Turcz. ex Ledeb.) Boeckeler (syn. Kobresia hyperborea A. E. Porsild). Species of Kobresia Willd. were recently transferred to Carex to make it a monophyletic genus (Global Carex Group, 2015).

Specimens Examined: Canada. Nunavut: Kitikmeot Region: Coppermine [Kugluktuk] [67°49′36″N, 115°5′36″W ± 1.5 km], 10 August 1962, J. A. Larsen s.n. (CAN-286320); flats on W side of Fockler Creek, above spruce forest in creek valley, ca. 2.2 km S of Sandstone Rapids, Coppermine River, 67°25′49″N, 115°37′55″W ± 50 m, 152 m, 1 July 2014, Saarela, Sokoloff & Bull 3118 (ALA, ALTA, CAN); E side of Fockler Creek, just above its confluence with Coppermine River, ca. 1.1 km SW of Sandstone Rapids, 67°26′30.6″N, 115°39′4.3″W ± 50 m, 135 m, 4 July 2014, Saarela, Sokoloff & Bull 3368 (CAN, NY, QFA, WIN); flats atop and upper slopes of Coppermine Mountains, N/W side of Coppermine River, 67°14′43.7″N, 115°38′51.2″W ± 150 m, 422 m, 9 9 July 2014, Saarela, Sokoloff & Bull 3746 (CAN, MT); Kugluk (Bloody Falls) Territorial Park, flats above boardwalk W of Bloody Falls, 67°44′34.5″N, 115°22′27″W ± 100 m, 135 m, 16 July 2014, Saarela, Sokoloff & Bull 4056 (CAN, MO, O); W of Kugluktuk on tundra flats above Coppermine River, S of 1 Coronation Drive and N of community power plant, 67°49′28.97″N, 115°5′0.2″W ± 100 m, 8 m, 22 July 2014, Saarela, Sokoloff & Bull 4235 (CAN, UBC, US).

Carex capillaris subsp. fuscidula (V. I. Krecz. ex T. V. Egorova) Á. Löve & D. Löve—Hair sedge | Circumpolar-alpine

Previously recorded from the study area (Porsild & Cody, 1980), but we were unable to locate a voucher specimen. We made collections at Fockler Creek, Kugluk (Bloody Falls) Territorial Park and Kugluktuk. Elsewhere in the Canadian Arctic recorded from Baffin, Banks, Ellesmere and Victoria islands, and a few sites on mainland Nunavut and Northwest Territories (Porsild & Cody, 1980; Aiken et al., 2007; Saarela et al., 2013a). The taxonomy of the Carex capillaris L. aggregate is reviewed in Saarela et al. (2013a).

Specimens Examined: Canada. Nunavut: Kitikmeot Region: Kugluktuk, rocky slopes of North Hill, 67°49′31.4″N, 115°6′54″W ± 100 m, 42 m, 29 June 2014, Saarela, Sokoloff & Bull 3081 (CAN); flats on W side of Fockler Creek, above spruce forest in creek valley, ca. 2.2 km S of Sandstone Rapids, Coppermine River, 67°25′49″N, 115°37′55″W ± 50 m, 152 m, 1 July 2014, Saarela, Sokoloff & Bull 3111 (CAN); old riverbed of Fockler Creek, ca. 2.3 km SSE of Sandstone Rapids, Coppermine River, 67°25′45.7″N, 115°37′21.8″W ± 25 m, 166 m, 1 July 2014, Saarela, Sokoloff & Bull 3162 (CAN, MO, MT, O, US); sedge meadow adjacent to small lake on flats N of Fockler Creek, ca. 1.5 km SSE of Sandstone Rapids, Coppermine River, 67°26′8.8″N, 115°37′35.9″W ± 20 m, 168 m, 2 July 2014, Saarela, Sokoloff & Bull 3227 (CAN, UBC); S of Fockler Creek, along small tributary that runs into Fockler Creek, ca. 2.3 km S of Sandstone Rapids, Coppermine River, 67°25′44.9″N, 115°38′25.9″W ± 100 m, 152 m, 3 July 2014, Saarela, Sokoloff & Bull 3259 (ALA, CAN); S of Fockler Creek, along small tributary that runs into Fockler Creek, ca. 2.3 km S of Sandstone Rapids, Coppermine River, 67°25′44.9″N, 115°38′25.9″W ± 100 m, 152 m, 3 July 2014, Saarela, Sokoloff & Bull 3278 (ALA, ALTA, CAN); Kugluk (Bloody Falls) Territorial Park, upper ledges of rocky (gabbro) S-facing cliffs above the start of Bloody Falls (W bank of river), just E of Portage Trail, 67°44′21.7″N, 115°22′42.2″W ± 25 m, 46 m, 14 July 2014, Saarela, Sokoloff & Bull 3944 (CAN, UBC); Kugluk (Bloody Falls) Territorial Park, flats above boardwalk W of Bloody Falls, 67°44′34.5″N, 115°22′27″W ± 100 m, 135 m, 16 July 2014, Saarela, Sokoloff & Bull 4061 (CAN, MO, MT, US); W of Kugluktuk on tundra flats above Coppermine River, S of 1 Coronation Drive and N of community power plant, 67°49′28.97″N, 115°5′0.2″W ± 100 m, 8 m, 22 July 2014, Saarela, Sokoloff & Bull 4237 (CAN, K, NY, QFA, WIN); W of Kugluktuk on tundra flats above Coppermine River, S of 1 Coronation Drive and N of power plant, 67°49′28.97″N, 115°5′0.2″W ± 100 m, 8 m, 25 July 2014, Saarela, Sokoloff & Bull 4370 (ALTA, CAN, O).

Carex capitata L., Fig. 21—Capitate sedge | Circumboreal-polar | Noteworthy Record

Figure 21 Carex capitata.

(A) Inflorescence, Saarela et al. 3574. (B) Habit, Saarela et al. 3515. Photographs by P. C. Sokoloff (A) and J. M. Saarela (B).

Our Subarctic collections, from Melville Creek and Kendall River, are the first records of this primarily boreal species for the study area and Nunavut, and represent a northwestern range extension from the nearest known locations along the eastern arm of Great Bear Lake (Porsild & Cody, 1980; Villaverde Hidalgo, 2012). This species and the closely related Carex arctogena Harry Sm. have been variously recognised taxonomically. Villaverde Hidalgo (2012) demonstrated they are distinct species. Previous records of Carex capitata from Nunavut (Gould & Walker, 1997; Blaney & Kotanen, 2001) are now referred to Carex arctogena (Villaverde Hidalgo, 2012). At the Melville Creek site it was locally common in a mesic meadow around a small pond with Rubus arcticus subsp. acaulis, Salix glauca and Betula glandulosa; at the Kendall River site it grew in mesic tundra with Betula glandulosa, Dasiphora fruticosa, Eriophorum angustifolium and Salix spp. White spruce was common at both sites.

Specimens Examined: Canada. Nunavut: Kitikmeot Region: confluence of Coppermine River and Melville Creek, just W of Coppermine Mountains, 67°15′52″N, 115°30′55.3″W ± 350 m, 178–190 m, 7 July 2014, Saarela, Sokoloff & Bull 3515 (ALA, CAN, UBC, WIN); confluence of Coppermine and Kendall rivers (NW side of Coppermine River, S side of Kendall River), 67°6′51.1″N, 116°8′18.3″W ± 150 m, 220 m, 7 July 2014, Saarela, Sokoloff & Bull 3574 (ALTA, CAN, MO, MT, O).

Carex chordorrhiza Ehrh. ex L. f.—Creeping sedge | Circumboreal-polar

Previously recorded from Kugluktuk (Porsild & Cody, 1980). We made collections of this strongly rhizomatous sedge at Fockler Creek, Kendall River and near Heart Lake growing in wetlands around small ponds. Elsewhere in the Canadian Arctic recorded from southern Baffin Island, Victoria Island and mainland sites (Porsild & Cody, 1980; Cody, Scotter & Zoltai, 1989; Aiken et al., 2007; Saarela et al., 2013a; Bennett, 2015).

Specimens Examined: Canada. Nunavut: Kitikmeot Region: Coppermine [Kugluktuk], vicinity of post [67°49′36″N, 115°5′36″W ± 1.5 km], 26 July 1949, A. E. Porsild 17162 (CAN-127542); ridge top N of Fockler Creek and S of Tundra Lake, ca. 3.8 km SE of Sandstone Rapids, Coppermine River, 67°25′18.8″N, 115°35′3.6″W ± 3 m, 258 m, 5 July 2014, Saarela, Sokoloff & Bull 3420 (CAN, UBC); confluence of Coppermine and Kendall rivers (NW side of Coppermine River, S side of Kendall River), small ponds on flats adjacent to Coppermine River, 67°6′44.7″N, 116°8′6.1″W ± 100 m, 213 m, 7 July 2014, Saarela, Sokoloff & Bull 3586 (ALA, CAN); ca. 0.5 km SW of Heart Lake, SW of Kugluktuk, 7.5 km SW of mouth of Coppermine River, 67°47′52″N, 115°14′14.4″W ± 350 m, 66 m, 23 July 2014, Saarela, Sokoloff & Bull 4279 (ALTA, CAN, O).

Carex concinna R. Br., Fig. 22—Low northern sedge, beauty sedge | North America (N) | Noteworthy Record

Figure 22 Carex concinna.

(A) Inflorescence, Saarela et al. 3567. (B) Habit, Saarela et al. 3569. Photographs by J. M. Saarela.

Newly recorded for the study area and a northeastern range extension from the nearest known sites along eastern Great Bear Lake (Porsild & Cody, 1980). We made collections at Fockler Creek, Big Creek, Kendall River, Escape Rapids and Kugluk (Bloody Falls) Territorial Park. In the study area this primarily boreal species grows in disturbed areas on slopes, ridge, along trails and in the understory of white spruce forest. Elsewhere in Nunavut known only from the Nueltin Lake area (Porsild, 1950b; Cody, 1978; Porsild & Cody, 1980) and Akimiski Island (Riley, 1981). It was recently recorded from the Arctic ecozone in the Brock River area of Northwest Territories and also known from sites west of there (Saarela et al., 2013a). The taxon was described from material collected by J. Richardson between Point Lake and the Arctic coast (i.e., along the Coppermine River, possibly including material gathered from the study area) and adjacent wooded country from latitude 54 to 64 degrees (Richardson, 1823).

Specimens Examined: Canada. Nunavut: Kitikmeot Region: slopes on E side of Coppermine River, N of its confluence with Fockler Creek, ca. 0.8 km SW of Sandstone Rapids, 67°26′36.9″N, 115°38′50.1″W ± 50 m, 128 m, 4 July 2014, Saarela, Sokoloff & Bull 3398 (CAN); forest and slopes at confluence of Big Creek and Coppermine River, N side of Coppermine River, S side of Coppermine Mountains, 67°14′29.3″N, 116°2′44.5″W ± 250 m, 180–199 m, 7 July 2014, Saarela, Sokoloff & Bull 3569 (CAN, K, MT, QFA, US, WIN); confluence of Coppermine and Kendall rivers (NW side of Coppermine River, S side of Kendall River), 67°6′51.1″N, 116°8′18.3″W ± 150 m, 220 m, 7 July 2014, Saarela, Sokoloff & Bull 3583 (CAN, UBC); S-facing slopes above Coppermine River and below spruce forest, ca. 7.8 km NNE of Sandstone Rapids, 67°31′16.2″N, 115°36′52.1″W ± 200 m, 110 m, 8 July 2014, Saarela, Sokoloff & Bull 3649 (CAN, MO, O, US); SE-facing slopes above Escape Rapids, W side of Coppermine River, 67°36′58.7″N, 115°29′18.3″W ± 99 m, 50 m, 8 July 2014, Saarela, Sokoloff & Bull 3734 (ALA, CAN); Kugluk (Bloody Falls) Territorial Park, rocky valley immediately SW of Bloody Falls, along rough marked section of Portage Trail, upper pond just W of Bloody Falls, 67°44′39.5″N, 115°22′28.9″W ± 10 m, 15 m, 13 July 2014, Saarela, Sokoloff & Bull 3896 (ALTA, CAN); Kugluk (Bloody Falls) Territorial Park, top of sandy ridge, ca. 0.75 km W of Bloody Falls., 67°44′45.7″N, 115°23′4.6″W ± 25 m, 56 m, 15 July 2014, Saarela, Sokoloff & Bull 4041 (CAN).

Carex fuliginosa subsp. misandra (R. Br.) Nyman—Short leaf sedge | Circumpolar-alpine

Previously recorded from the study area, as Carex misandra R. Br. (Porsild & Cody, 1980), but we were unable to locate a voucher specimen. We made collections at Fockler Creek, Kugluk (Bloody Falls) Territorial Park and Kugluktuk. Taxonomy follows Elven et al. (2011) and is reviewed in Saarela et al. (2013a). Widespread throughout the Canadian Arctic (Porsild & Cody, 1980; Cody, Scotter & Zoltai, 1989; Korol, 1992; Aiken et al., 2007; Saarela et al., 2013a).

Specimens Examined: Canada. Nunavut: Kitikmeot Region: E side of Fockler Creek, in valley just above creek’s confluence with the Coppermine River, ca. 1.4 km SSW of Sandstone Rapids, 67°26′14.5″N, 115°38′34.8″W ± 50 m, 146 m, 4 July 2014, Saarela, Sokoloff & Bull 3355 (ALA, ALTA, CAN, MO, O); Kugluk (Bloody Falls) Territorial Park, upper ledges of rocky (gabbro) S-facing cliffs above the start of Bloody Falls (W bank of River), just E of Portage Trail, 67°44′21.7″N, 115°22′42.2″W ± 25 m, 46 m, 14 July 2014, Saarela, Sokoloff & Bull 3947 (CAN, UBC); W of Kugluktuk on tundra flats above Coppermine River, S of 1 Coronation Drive and N of power plant, 67°49′28.97″N, 115°5′0.2″W ± 100 m, 8 m, 25 July 2014, Saarela, Sokoloff & Bull 4376 (CAN).

Carex glacialis Mack.—Glacier sedge | Circumpolar–alpine

Previously recorded from Kugluktuk (Porsild & Cody, 1980). We collected it there on North Hill, and also at Fockler Creek and Kugluk (Bloody Falls) Territorial Park. Elsewhere in the Canadian Arctic recorded from Baffin, Coats, Ellesmere, Nottingham and Victoria islands, and numerous mainland sites (Porsild & Cody, 1980; Aiken et al., 2007; Saarela et al., 2013a).

Specimens Examined: Canada. Nunavut: Kitikmeot Region: Coppermine [Kugluktuk], vicinity of post [67°49′36″N, 115°5′36″W ± 1.5 km], 26 July 1949, A. E. Porsild 17163 (CAN-12748); Kugluktuk, flat mesa at top of North Hill, 67°49′32″N, 115°6′39″W ± 100 m, 50 m, 29 June 2014, Saarela, Sokoloff & Bull 3088 (CAN, UBC); old riverbed of Fockler Creek, ca. 2.3 km SSE of Sandstone Rapids, Coppermine River, 67°25′45.7″N, 115°37′21.8″W ± 25 m, 166 m, 1 July 2014, Saarela, Sokoloff & Bull 3165 (CAN); third ridge N of Fockler Creek overlooking small lake, ca. 1.7 km SSE of Sandstone Rapids, Coppermine River, 67°26′7.3″N, 115°37′28″W ± 3 m, 171 m, 2 July 2014, Saarela, Sokoloff & Bull 3222 (ALA, ALTA, CAN, O); Kugluk (Bloody Falls) Territorial Park, flats on top of mountain on W side of Coppermine River, just S of the start of Bloody Falls Rapids, 67°44′2.8″N, 115°23′39.3″W ± 250 m, 110 m, 14 July 2014, Saarela, Sokoloff & Bull 3998 (CAN, MO, MT, US).

Carex glareosa Wahlenb. subsp. glareosa—Gravel sedge | Circumpolar | Noteworthy Record

Our two collections of this species, which grows on tidal flats and in brackish marshes, are the first records for the study area. We encountered it at Richardson Bay and on an island at the mouth of the Coppermine River. Elsewhere in Nunavut known from the Bathurst Inlet area, the western shore of Hudson Bay, and Baffin, Coats, Devon and Ellesmere islands (Porsild & Cody, 1980; Cody, Scotter & Zoltai, 1984, 1989; Korol, 1992; Aiken et al., 2007). In Northwest Territories known from the Mackenzie Delta area, Ulukhaktok (Victoria Island) and the Brock Lagoon (Porsild & Cody, 1980; Saarela et al., 2013a). Porsild & Cody (1980) treated it as Carex glareosa var. amphigena Fernald, considered a synonym of subsp. glareosa by Toivonen (2002). The other infraspecific taxon, subsp. pribylovensis (Macoun) G. Halliday & Chater, is known only from the Pribilof and Aleutian islands, Alaska (Toivonen, 2002).

Specimens Examined: Canada. Nunavut: Kitikmeot Region: Richardson Bay, confluence of Richardson and Rae rivers at Coronation Gulf, ca. 20 km WNW of Kugluktuk, 67°54′11.2″N, 115°32′27.4″W ± 200 m, 0 m, 8 July 2014, Saarela, Sokoloff & Bull 3687 (CAN, MO, MT, QFA, US, WIN); unnamed island just E (ca. 3.3 km) of Kugluktuk at mouth of Coppermine River, 67°49′29.2″N, 115°1′3.2″W ± 50 m, 1 m, 8 July 2014, Saarela, Sokoloff & Bull 3727 (ALA, ALTA, CAN, O, UBC).

Carex gynocrates Wormsk. ex Drejer, Figs. 23A and 23B—Northern bog sedge | Asian (NE)–amphi-Beringian–North American (N) | Noteworthy Record

Figure 23 Carex gynocrates, Carex krausei, and Carex lachenalii.

Carex gynocrates: (A) inflorescence, Saarela et al. 3276. (B) Habit, Saarela et al. 3276. Carex krausei: (C) habit, Saarela et al. 3856. Carex lachenalii: (D) habit, vicinity of Fockler Creek, Nunavut, 2 July 2014. Photographs by R. D. Bull (A, B) and J. M. Saarela (C, D).

Our two collections of this calciphile are the first records for the study area and mainland Nunavut, and represent a northeastern range extension from the nearest known sites along the eastern shore of Great Bear Lake (Porsild & Cody, 1980). In Kugluk (Bloody Falls) Territorial Park, we encountered it in a large wet silty/sandy meadow with Eriophorum triste, Carex fuliginosa subsp. misandra, Carex membranacea, Carex microglochin, Carex simpliciuscula, Eriophorum callitrix, Eriophorum triste and Saxifraga aizoides. Near Fockler Creek it grew in a wet sedge meadow with Carex membranacea and Trichophorum cespitosum. Elsewhere in Nunavut this primarily boreal species is known from Baffin Island (Porsild & Cody, 1980; Aiken et al., 2007).

Specimens Examined: Canada. Nunavut: Kitikmeot Region: S of Fockler Creek, along small tributary that runs into Fockler Creek, ca. 2.3 km S of Sandstone Rapids, Coppermine River, 67°25′44.9″N, 115°38′25.9″W ± 100 m, 152 m, 3 July 2014, Saarela, Sokoloff & Bull 3276 (ALA, CAN, MT, UBC); Kugluk (Bloody Falls) Territorial Park, wet meadow between Coppermine River and large sand hills on W side of river, 0.5 km W of Bloody Falls, 67°44′44.8″N, 115°22′48.3″W ± 15 m, 33 m, 15 July 2014, Saarela, Sokoloff & Bull 4048 (CAN).

Carex holostoma Drejer—Arctic marsh sedge | Circumpolar?

Previously recorded along the Coppermine River (Porsild & Cody, 1980). We collected it at Fockler Creek, Kugluk (Bloody Falls) Territorial Park, Heart Lake and the Kugluktuk sewage retention pond. At the latter, well-fertilised site, the plants were extremely vigorous and the vegetation very lush. The study area is the known eastern limit of the western half of this species’ range in Canada. There is a large gap in the Central Arctic (Porsild & Cody, 1980), which is partly closed by record(s) from the Hood River (Gould & Walker, 1997). Elsewhere in the Canadian Arctic recorded from southern Baffin Island, Victoria and Southampton islands, as well as a few mainland sites (Porsild & Cody, 1980; Aiken et al., 2007; Saarela et al., 2013a).

Specimens Examined: Canada. Nunavut: Kitikmeot Region: Coppermine River, 10 August 1962, J. A. Larsen s.n. (CAN-287133); flats on W side of Fockler Creek, above spruce forest in creek valley, ca. 2.2 km S of Sandstone Rapids, Coppermine River, 67°25′49″N, 115°37′55″W ± 50 m, 152 m, 1 July 2014, Saarela, Sokoloff & Bull 3115 (CAN, UBC); Kugluk (Bloody Falls) Territorial Park, upper ledges of rocky (gabbro) S-facing cliffs above the start of Bloody Falls (W bank of River), just E of Portage Trail, 67°44′21.7″N, 115°22′42.2″W ± 25 m, 46 m, 14 July 2014, Saarela, Sokoloff & Bull 3943 (ALA, CAN); ca. 0.5 km SW of Heart Lake, SW of Kugluktuk, 7.5 km SW of mouth of Coppermine River, 67°47′52″N, 115°14′14.4″W ± 350 m, 66 m, 23 July 2014, Saarela, Sokoloff & Bull 4309 (ALTA, CAN, O); creek just N of sewage retention pond (used as sewage outlet), 5.1 km SW of Coppermine River, 67°48′59.1″N, 115°12′5.8″W ± 25 m, 34 m, 23 July 2014, Saarela, Sokoloff & Bull 4334 (CAN, MO, MT).

Carex krausei Boeckeler, Fig. 23C—Krause’s sedge | Circumpolar–alpine | Noteworthy Record

Newly recorded for the study area, closing a distribution gap between sites at Bathurst Inlet (Porsild & Cody, 1980, as Carex capillaris subsp. robustior (Lange) Böcher) (Bennett, 2015), Hood River (Gould & Walker, 1997) and Tuktut Nogait National Park and vicinity (Saarela et al., 2013a). We made collections at Fockler Creek, Melville Creek, Kugluk (Bloody Falls) Territorial Park and Kugluktuk. Elsewhere in the Canadian Arctic recorded from southern Baffin Island and Banks, Coats, Ellesmere and Southampton islands (Aiken et al., 2007). Taxonomy follows Elven et al. (2011). Taxonomy of the Carex capillaris aggregate (including Carex krausei) is reviewed in Saarela et al. (2013a).

Specimens Examined: Canada. Nunavut: Kitikmeot Region: E side of Fockler Creek, in valley just above creek’s confluence with the Coppermine River, ca. 1.4 km SSW of Sandstone Rapids, 67°26′14.5″N, 115°38′34.8″W ± 50 m, 146 m, 4 July 2014, Saarela, Sokoloff & Bull 3352 (CAN); E side of Fockler Creek, just above its confluence with Coppermine River, ca. 1.1 km SW of Sandstone Rapids, 67°26′30.6″N, 115°39′4.3″W ± 50 m, 135 m, 4 July 2014, Saarela, Sokoloff & Bull 3369 (CAN, MO, MT, US); confluence of Coppermine River and Melville Creek, just W of Coppermine Mountains, 67°15′52″N, 115°30′55.3″W ± 350 m, 178–190 m, 7 July 2014, Saarela, Sokoloff & Bull 3494 (CAN, UBC); Kugluk (Bloody Falls) Territorial Park, rocky valley immediately SW of Bloody Falls, along rough marked section of Portage Trail, 67°44′34″N, 115°22′16″W ± 50 m, 20 m, 13 July 2014, Saarela, Sokoloff & Bull 3856 (CAN, NY, QFA, WIN); Kugluk (Bloody Falls) Territorial Park, small rocky meadow along small stream that runs into Coppermine River just below Bloody Falls, about 1 km W of Bloody Falls, 67°44′40.1″N, 115°23′37.5″W ± 15 m, 48 m, 15 July 2014, Saarela, Sokoloff & Bull 4029 (ALTA, CAN, O); W of Kugluktuk on tundra flats above Coppermine River, S of 1 Coronation Drive and N of community power plant, 67°49′28.97″N, 115°5′0.2″W ± 100 m, 8 m, 22 July 2014, Saarela, Sokoloff & Bull 4272 (ALA, CAN).

Carex lachenalii Schkuhr, Fig. 23D—Short leaf sedge | Circumpolar-alpine | Noteworthy Record

Newly recorded for the study area, and a northeastern range extension in the central Canadian Arctic. We made collections at Fockler Creek (snowbed habitat), Coppermine Mountains (on a siksik den) and Kugluktuk (snowbed community on east-facing slope at base of cliff, mossy ground). The nearest collection is from northern Great Bear Lake (Porsild & Cody, 1980). Elsewhere in Nunavut recorded from Hood River (Gould & Walker, 1997), the eastern mainland, southern Baffin Island, and Coats, Southampton and Victoria islands (Porsild & Cody, 1980; Aiken et al., 2007). There is a large distribution gap in the Central Arctic similar to the distribution of Carex holostoma (Porsild & Cody, 1980). It is not known if this gap is real, or an artefact of sampling. Plants recognised as Carex lachenalii in Saarela et al. (2013a) are in fact Carex marina Dewey, as noted on the last page of that publication; Carex lachenalii is not known from that area.

Specimens Examined: Canada. Nunavut: Kitikmeot Region: NW-facing slope above tributary of Fockler Creek, ca. 2.4 km SSW of Sandstone Rapids, Coppermine River, 67°25′46″N, 115°38′49.4″W ± 50 m, 149 m, 3 July 2014, Saarela, Sokoloff & Bull 3295 (ALA, CAN, UBC); flats atop and upper slopes of Coppermine Mountains, N/W side of Coppermine River, 67°14′53.6″N, 115°38′37.9″W ± 15 m, 401 m, 9 9 July 2014, Saarela, Sokoloff & Bull 3758 (CAN); SE edge of Kugluktuk, rocky cliffs overlooking Coppermine River, 67°49′9.2″N, 115°5′40.4″W ± 50 m, 28 m, 24 July 2014, Saarela, Sokoloff & Bull 4348 (ALA, ALTA, CAN, O).

Carex livida (Wahlenb.) Willd.—Livid sedge | European (N)–Asian (NW) & Asian Pacific & North American | Noteworthy Record

Our single Subarctic collection is the first record for mainland Nunavut. The colonial, rhizomatous plants were growing in a large, wet, calcareous meadow with Carex microglochin and Eriophorum angustifolium near Fockler Creek. Rothrock & Reznicek (2002) reported this species as having a “very scattered” distribution. Elsewhere in Nunavut recorded from Akimiski (Riley, 1981; Blaney & Kotanen, 2001) and Stromness islands (Cayouette & Darbyshire J86-237, DAO-681352) in James Bay. In northern Quebec and Labrador recorded as “not extending into the Arctic zone” (Cayouette, 2008). In Northwest Territories known from the Mackenzie Delta area, where one collection is within the Arctic ecozone (Eskimo Lake Basin, 69°N, 132°30′W, 18 August 1927, Porsild & Porsild 2965, CAN-21930, ALTA-29871; Porsild, 1943), the west end of Great Slave Lake (Porsild & Cody, 1980), Heart Lake (Cody & Talbot, 1978), Aubry Lake (Riewe & Marsh 343, CAN-433228), a site along the Mackenzie River southwest of Great Bear Lake (DeCarlo & Kershaw s.n., CAN-588721), the Ebbutt Hills (Reid 1344, ALTA-78494), Scotty Creek (Garon-Labrecque et al., 2015 [2016]) and Nahanni National Park (Bennett, 2013). Carex livida was treated as a rare plant in Yukon (Douglas et al., 1981), but not in Northwest Territories (McJannet, Argus & Cody, 1995), although Douglas et al. (1981) earlier noted it to be rare there. It was recorded as “Sensitive” by the Working Group on General Status of NWT Species (2011). A collection from northern Yukon (Firth River, 68.88646°N, 140.44214°W, 395 m, 19 July 2008, Bennett, Demers, Touzi & Reimer 08-375, CAN-590238) is from a non-Arctic area. Porsild & Cody (1980) recognised it as Carex livida var. grayana (Dewey) Fernald, a name Elven et al. (2011) considered to be a synonym of Carex livida (no infraspecific taxa recognised) and that Rothrock & Reznicek (2002) did not treat.

Specimens Examined: Canada. Nunavut: Kitikmeot Region: flats N of Fockler Creek, between Fockler Creek and Sleigh Creek, ca. 0.6 km SE of Sandstone Rapids, Coppermine River, 67°26′26″N, 115°37′40.7″W ± 3 m, 125 m, 6 July 2014, Saarela, Sokoloff & Bull 3485 (ALA, CAN, O, UBC).

Carex marina Dewey, Fig. S3A—Sea sedge | Circumpolar-alpine

Previously recorded from Kugluktuk (Porsild & Cody, 1980). We made numerous collections at Fockler Creek, in Kugluk (Bloody Falls) Territorial Park, and near the Kugluktuk sewage retention pond. In the Canadian Arctic recorded from Baffin, Banks, Ellesmere, Nottingham, Southampton and Victoria islands, as well as mainland sites (Porsild & Cody, 1980; Aiken et al., 2007; Saarela et al., 2013a). Elven et al. (2011) provisionally accepted subsp. marina and subsp. pseudolagopina (T. J. Sørensen) Böcher, with only the former present in North America, whereas Toivonen (2002) did not recognise subspecies. We follow the latter authority. Porsild & Cody (1980) recognised the taxon as Carex amblyorhyncha V. I. Krecz., a name now treated as a synonym of Carex marina (Toivonen, 2002; Elven et al., 2011).

Specimens Examined: Canada. Nunavut: Kitikmeot Region: Coppermine [Kugluktuk] [67°49′36″N, 115°5′36″W ± 1.5 km], 10 August 1962, J. A. Larsen s.n. (CAN-286311); S of Fockler Creek, along small tributary that runs into Fockler Creek, ca. 2.3 km S of Sandstone Rapids, Coppermine River, 67°25′44.9″N, 115°38′25.9″W ± 100 m, 152 m, 3 July 2014, Saarela, Sokoloff & Bull 3261 (CAN, UBC); tundra below Tundra Lake and Fockler Creek, ca. 4.1 km SE of Sandstone Rapids, Coppermine River, 67°25′20.7″N, 115°34′17.2″W ± 25 m, 271 m, 5 July 2014, Saarela, Sokoloff & Bull 3428 (ALA, ALTA, CAN); small unnamed tributary of Sleigh Creek, just upstream from its confluence with Sleigh Creek, 67°26′43.9″N, 115°36′57″W ± 5 m, 156 m, 6 July 2014, Saarela, Sokoloff & Bull 3463 (CAN, MO, MT, O); Kugluk (Bloody Falls) Territorial Park, rocky valley immediately SW of Bloody Falls, along rough marked section of Portage Trail, 67°44′34″N, 115°22′16″W ± 50 m, 20 m, 13 July 2014, Saarela, Sokoloff & Bull 3859 (CAN); Kugluk (Bloody Falls) Territorial Park, flats 1 km W of Bloody Falls, W side of Coppermine River, 67°44′32.6″N, 115°23′23.1″W ± 10 m, 70 m, 16 July 2014, Saarela, Sokoloff & Bull 4095 (ALA, ALTA, CAN, K, MO, MT, NY, UBC); Kugluk (Bloody Falls) Territorial Park, rocky sand beach just below Bloody Falls, W side of Coppermine River, vicinity of confluence with small creek, beach seasonally flooded, 67°44′54.5″N, 115°22′17.2″W ± 75 m, 9 m, 17 July 2014, Saarela, Sokoloff & Bull 4121 (CAN); creek just N of sewage retention pond (used as sewage outlet), 5.1 km SW of Coppermine River, 67°48′59.1″N, 115°12′5.8″W ± 25 m, 34 m, 23 July 2014, Saarela, Sokoloff & Bull 4332 (CAN, K, MO, O, QFA, US, WIN).

Carex maritima Gunnerus, Figs. S3B and S3C—Maritime sedge | Circumpolar-alpine

Previously recorded from Kugluktuk (Porsild & Cody, 1980). We made collections of this strongly rhizomatous sedge at Richardson Bay, Kugluk (Bloody Falls) Territorial Park and Heart Lake. Elsewhere in the Canadian Arctic recorded from Baffin, Banks, Coats, Devon, Ellesmere, Southampton and Victoria islands, as well as mainland sites (Porsild & Cody, 1980; Cody, Scotter & Zoltai, 1989; Korol, 1992; Cody, Reading & Line, 2003; Aiken et al., 2007; Saarela et al., 2013a).

Specimens Examined: Canada. Nunavut: Kitikmeot Region: Coppermine [Kugluktuk] [67°49′36″N, 115°5′36″W ± 1.5 km], 2 August 1962, J. A. Larsen s.n. (CAN-287134); Richardson Bay, confluence of Richardson and Rae rivers at Coronation Gulf, ca. 20 km WNW of Kugluktuk, 67°54′11.2″N, 115°32′27.4″W ± 200 m, 0 m, 8 July 2014, Saarela, Sokoloff & Bull 3686 (CAN, UBC); Kugluk (Bloody Falls) Territorial Park, rocky sand beach just below Bloody Falls, W side of Coppermine River, vicinity of confluence with small creek, beach seasonally flooded, 67°44′54.5″N, 115°22′17.2″W ± 75 m, 9 m, 17 July 2014, Saarela, Sokoloff & Bull 4117 (ALA, ALTA, CAN); Heart Lake, SW of Kugluktuk, 6.4 km SW of mouth of Coppermine River, 67°48′7.8″N, 115°13′22.7″W ± 350 m, 33 m, 23 July 2014, Saarela, Sokoloff & Bull 4295 (CAN, MO, O).

Carex membranacea Hook., Fig. S4—Fragile sedge | Amphi-Beringian–North America (N)

Previously recorded from Kugluktuk (Porsild & Cody, 1980). We made collections at Fockler Creek and Kugluk (Bloody Falls) Territorial Park. Widespread throughout the Canadian Arctic (Porsild & Cody, 1980; Cody, Scotter & Zoltai, 1989; Korol, 1992; Aiken et al., 2007; Saarela et al., 2013a).

Specimens Examined: Canada. Nunavut: Kitikmeot Region: Coppermine [Kugluktuk] [67°49′36″N, 115°5′36″W ± 1.5 km], 2 August 1962, J. A. Larsen s.n. (CAN-287134); old riverbed of Fockler Creek, ca. 2.3 km SSE of Sandstone Rapids, Coppermine River, 67°25′45.7″N, 115°37′21.8″W ± 25 m, 166 m, 1 July 2014, Saarela, Sokoloff & Bull 3161 (ALTA, CAN, O); S of Fockler Creek, along small tributary that runs into Fockler Creek, ca. 2.3 km S of Sandstone Rapids, Coppermine River, 67°25′44.9″N, 115°38′25.9″W ± 100 m, 152 m, 3 July 2014, Saarela, Sokoloff & Bull 3265 (CAN, MO, MT); gravel bar in Fockler Creek just above its confluence with Coppermine River, ca. 1.4 km SSW of Sandstone Rapids, 67°26′21.4″N, 115°38′54″W ± 5 m, 141 m, 4 July 2014, Saarela, Sokoloff & Bull 3363 (CAN, UBC); Kugluk (Bloody Falls) Territorial Park, small rocky meadow along small stream that runs into Coppermine River just below Bloody Falls, about 1 km W of Bloody Falls, 67°44′40.1″N, 115°23′37.5″W ± 15 m, 48 m, 15 July 2014, Saarela, Sokoloff & Bull 4028 (ALA, CAN).

Carex microglochin Wahlenb.—Bristle sedge | American Beringian–North American–amphi-Atlantic–European (N/C) & Asian (C), bipolar | Noteworthy Record

Newly recorded for the study area, closing a distribution gap between Bathurst Inlet, the eastern arm of Great Bear Lake and Tuktut Nogait National Park and vicinity (Porsild, 1943; Porsild & Cody, 1980; Saarela et al., 2013a). We made collections at Fockler Creek, Big Creek, Kugluk (Bloody Falls) Territorial Park and Kugluktuk. The species was uncommon to locally common in wet areas, including sedge meadows, river beds, disturbed clay flats and silty/sandy meadows, growing with such species as Carex aquatilis subsp. stans, Carex fuliginosa subsp. misandra, Carex membranacea, Carex simpliciuscula subsp. subholarctica, Eriophorum callitrix, Eriophorum triste, Pinguicula vulgaris, Saxifraga aizoides, Tofieldia pusilla and Trichophorum cespitosum. Elsewhere in the Canadian Arctic recorded from southern Baffin Island, Banks and Victoria islands, and a few other mainland sites (Porsild & Cody, 1980; Aiken et al., 2007; Saarela et al., 2013a). Taxonomy is reviewed in Saarela et al. (2013a).

Specimens Examined: Canada. Nunavut: Kitikmeot Region: S of Fockler Creek, along small tributary that runs into Fockler Creek, ca. 2.3 km S of Sandstone Rapids, Coppermine River, 67°25′44.9″N, 115°38′25.9″W ± 100 m, 152 m, 3 July 2014, Saarela, Sokoloff & Bull 3267 (CAN, UBC); forest and slopes at confluence of Big Creek and Coppermine River, N side of Coppermine River, S side of Coppermine Mountains, 67°14′29.3″N, 116°2′44.5″W ± 250 m, 180–199 m, 7 July 2014, Saarela, Sokoloff & Bull 3561 (ALA, ALTA, CAN); Kugluk (Bloody Falls) Territorial Park, rocky valley immediately SW of Bloody Falls, along rough marked section of Portage Trail, upper pond just W of Bloody Falls, 67°44′39.5″N, 115°22′28.9″W ± 10 m, 15 m, 13 July 2014, Saarela, Sokoloff & Bull 3895 (CAN); Kugluk (Bloody Falls) Territorial Park, wet meadow between Coppermine River and large sand hills on W side of river, 0.5 km W of Bloody Falls, 67°44′44.8″N, 115°22′48.3″W ± 15 m, 33 m, 15 July 2014, Saarela, Sokoloff & Bull 4047 (CAN, MO, O); W of Kugluktuk on tundra flats above Coppermine River, S of 1 Coronation Drive and N of community power plant, 67°49′28.97″N, 115°5′0.2″W ± 100 m, 8 m, 22 July 2014, Saarela, Sokoloff & Bull 4267 (CAN, MT, WIN).

Carex myosuroides Vill.—Mouse-tail bog sedge | Circumpolar-alpine

Previously recorded from Kugluktuk (Porsild & Cody, 1980) and previously recognised as Kobresia myosuroides (Vill.) Fiori (Global Carex Group, 2015). We made collections at Fockler Creek, Big Creek, Kugluk (Bloody Falls) Territorial Park and Kugluktuk. Elsewhere in the Canadian Arctic recorded from Baffin, Banks, Devon, Ellesmere, Melville, Somerset, Southampton and Victoria islands, as well as a few other mainland sites (Porsild & Cody, 1980; Aiken et al., 2007; Saarela et al., 2013a; Global Carex Group, 2015).

Specimens Examined: Canada. Nunavut: Kitikmeot region: Coppermine [Kugluktuk], Coronation Gulf, at mouth of Coppermine River, back of the village [67.822146°N, 115.078387°W ± 0.5 km], 4 August 1948, H. T. Shacklette 3284 (CAN-199943); SW-facing slope above (N side) of Fockler Creek, ca. 3.2 km SEK of Sandstone Rapids, Coppermine River, 67°25′26.2″N, 115°36′14″W ± 25 m, 193 m, 5 July 2014, Saarela, Sokoloff & Bull 3416 (ALA, CAN, UBC); forest and slopes at confluence of Big Creek and Coppermine River, N side of Coppermine River, S side of Coppermine Mountains, 67°14′29.3″N, 116°2′44.5″W ± 250 m, 180–199 m, 7 July 2014, Saarela, Sokoloff & Bull 3554 (CAN); Kugluk (Bloody Falls) Territorial Park, flats on top of mountain on W side of Coppermine River, just S of the start of Bloody Falls Rapids, 67°44′2.8″N, 115°23′39.3″W ± 250 m, 110 m, 14 July 2014, Saarela, Sokoloff & Bull 3991 (ALTA, CAN, O); W of Kugluktuk on tundra flats above Coppermine River, S of 1 Coronation Drive and N of power plant, 67°49′28.97″N, 115°5′0.2″W ± 100 m, 8 m, 25 July 2014, Saarela, Sokoloff & Bull 4372 (CAN, MO, MT).

Carex nardina Fr.—Nard sedge | Amphi-Beringian–North American–amphi-Atlantic (W) | Noteworthy Record

Newly recorded for the study area. We made collections at Kugluktuk, Fockler Creek, and Kugluk (Bloody Falls) Territorial Park. The nearest collections are from Great Bear Lake, Tuktut Nogait Park and vicinity including adjacent Nunavut, and Bathurst Inlet and vicinity (Porsild & Cody, 1980; Cody, Scotter & Zoltai, 1984; Gould & Walker, 1997; Cody & Reading, 2005; Saarela et al., 2013a). Elsewhere in the Canadian Arctic recorded from Baffin, Banks, Devon, Ellesmere, Mansel, Melville, Nottingham, Southampton and Victoria islands, as well as mainland sites (Porsild & Cody, 1980; Cody, Scotter & Zoltai, 1989; Cody & Reading, 2005; Aiken et al., 2007; Saarela et al., 2013a). Varying infraspecific taxonomic treatments have been proposed, as reviewed in Saarela et al. (2013a). A recent molecular and morphological study of the Carex nardina complex concluded the taxon is best recognised as a single variable species (Sawtell, 2012), as treated by Murray (2002a).

Specimens Examined: Canada. Nunavut: Kitikmeot region: Kugluktuk, flat mesa at top of North Hill, 67°49′32″N, 115°6′39″W ± 100 m, 50 m, 29 June 2014, Saarela, Sokoloff & Bull 3087 (ALA, ALTA, CAN); old riverbed of Fockler Creek, ca. 2.3 km SSE of Sandstone Rapids, Coppermine River, 67°25′45.7″N, 115°37′21.8″W ± 25 m, 166 m, 1 July 2014, Saarela, Sokoloff & Bull 3163 (CAN, UBC); SW-facing slope above (N side) of Fockler Creek, ca. 3.2 km SE of Sandstone Rapids, Coppermine River, 67°25′26.2″N, 115°36′14″W ± 25 m, 193 m, 5 July 2014, Saarela, Sokoloff & Bull 3413 (CAN, NY, QFA, US, WIN); Kugluk (Bloody Falls) Territorial Park, flats on top of mountain on W side of Coppermine River, just S of the start of Bloody Falls Rapids, 67°44′2.8″N, 115°23′39.3″W ± 250 m, 110 m, 14 July 2014, Saarela, Sokoloff & Bull 3997 (CAN, MO, MT, O).

Carex norvegica Retz.—Norway sedge | North American (NE)–amphi-Atlantic–Eurasian | Noteworthy Record

Our collections are the first for the study area and western Nunavut. We made one collection at Fockler Creek, where the species was uncommon on hummocks in mesic tundra, growing with Betula glandulosa, Carex scirpoidea subsp. scirpoidea, Carex vaginata and Vaccinium uliginosum. A second collection was made further north in Kugluk (Bloody Falls) Territorial Park in low shrub tundra, where the species was locally common, growing with Betula glandulosa, Carex bigelowii, Eriophorum vaginatum, Rhododendron lapponicum, Rumex arcticus and Salix spp.

The more northerly distributed Carex norvegica and the circumboreal Carex media R. Br. are morphologically similar and closely related (Gebauer, Röser & Hoffmann, 2015). They are sometimes considered synonyms (usually with Carex media treated as an infraspecific taxon of Carex norvegica), but are recognised as species in recent North American literature (Porsild & Cody, 1980; Murray, 2002b; Elven et al., 2011). Carex media was only recently reported for Nunavut, from the southeastern mainland (Cody & Reading, 2005). We initially had difficulties distinguishing Carex norvegica and Carex media, due to problems in published keys. In Porsild & Cody (1980), both taxa key out in the Eucarices tristigmatae key under the first lead, “terminal spike staminate”. This is an error as both species have terminal spikes gynecandrous, as noted correctly in the descriptions in that treatment. There is also a problem in the Flora of North America. The first clauses of the key in Murray (2002b) distinguishing Carex norvegica and Carex media (lead 13) are erroneously switched with respect to the taxon descriptions in the same publication, which are correct, and some useful distinguishing characters listed in the descriptions are not included in the key (e.g., perigynia length). A revised key is presented below. Our collection no. 4054 has perigynia 2.2–2.4 mm long, obovate, and mostly dark brown (a few are greenish becoming dark brown) and faintly nerved proximally. Pistillate scales are slightly shorter than the perigynia with distinctly hyaline margins to near their bases. The plants were growing in dense clumps, with culms up to 30 cm (current year culms shorter). No. 3220 are younger plants with green perigynia, but otherwise similar to no. 4054. These characters place our collections under Carex norvegica.

There are conflicting reports in the literature regarding the distribution of Carex norvegica. It was reported by Murray (2002b) as having an amphi-Atlantic distribution and reaching its eastern limit along western Hudson Bay, from where there are several records (Porsild & Cody, 1980; Cody, Scotter & Zoltai, 1989), whereas Elven et al. (2011) reported the broader distribution given above (also see maps in Kalela, 1944). In northwestern Canada it is recorded from a single Yukon collection (Ogilvie Mts.: river flats along Dempster Rd., mile 57–58, elev. 2,500–4,200 ft., in meadows between old channels, 23 July 1966, R. T. Porsild 294, CAN-303347, det. A. E. Porsild s.d., and A. A. Reznicek, 2006). This collection was also determined by T. Spribille, in 1997, as Carex norvegica cf. subsp. conicorostrata Kalela. The Yukon specimen was mapped for Carex norvegica in Porsild & Cody (1980) and Cody (2000), but Murray (2002b) did not include Yukon as part of its distribution. Elven et al. (2011) recognised three infraspecific taxa in Carex norvegica, following Kalela (1944): subsp. norvegica from Europe and Greenland, subsp. inserrulata Kalela from Canada and Greenland and subsp. conicorostrata from Russia and possibly Alaska. Murray (2002b) did not recognise infraspecific taxa, but noted further study of variation is needed; we follow his treatment. The reproductive characters in the Yukon collection match our specimens, and the Yukon plant is similarly densely cespitose (but much larger with culms up to 36 cm tall). Whether the Coppermine River plants are more closely related to the western or eastern populations is an open question. There remains a large and conspicuous distribution gap on mainland Nunavut between the study area and western Hudson Bay. Elsewhere in the Canadian Arctic recorded from southern Baffin Island, Southampton Island, and adjacent northern Quebec and Labrador (Aiken et al., 2007).

Key to distinguish Carex media and Carex norvegica, modified from Murray (2002b): 1. Perigynia green becoming dark brown or purple-black, veinless, elliptic or obovate, 2–2.5 mm long; pistillate scales dark brown or black, margins hyaline, equalling or shorter than perigynia; beak 0.2–0.3 mm longCarex norvegica

1′ Perigynia pale green becoming golden brown, often veined proximally, ovate, 2.5–3.5 mm long; pistillate scales dark brown to black to margins or distally hyaline, shorter than (often ½ length) perigynia; beak 0.3–0.4 mm longCarex media

Specimens Examined: Canada. Nunavut: Kitikmeot region: just above second ridge N of Fockler Creek, ca. 1.9 km SSE of Sandstone Rapids, Coppermine River, 67°26′3.3″N, 115°37′25.6″W ± 5 m, 169 m, 2 July 2014, Saarela, Sokoloff & Bull 3220 (CAN, UBC); Kugluk (Bloody Falls) Territorial Park, flats above boardwalk W of Bloody Falls, 67°44′34.5″N, 115°22′27″W ± 100 m, 135 m, 16 July 2014, Saarela, Sokoloff & Bull 4054 (ALA, ALTA, CAN, MO, O).

Carex petricosa Dewey subsp. petricosa, Fig. 24—Rock-dwelling sedge | Amphi-Beringian–Cordilleran & North America (NE) | Noteworthy Record

Figure 24 Carex petricosa subsp.

(A) Inflorescence, Saarela et al. 4173. (B) Habitat, Saarela et al. 3553. Photographs by J. M. Saarela.

Newly recorded for the study area. We made collections at six sites spanning the Subarctic and Arctic regions, but did not find it further north than Kugluk (Bloody Falls) Territorial Park. It grows in disturbed places on slopes, ridges, sand hills and mud boils on tundra flats, and is usually locally common when present. Associated species in the study area include Arnica angustifolia, Artemisia borealis subsp. borealis, Betula glandulosa, Bistorta vivipara, Calamagrostis purpurascens, Carex scirpoidea subsp. scirpoidea, Dasiphora fruticosa, Dryas integrifolia, Elymus alaskanus, Festuca rubra subsp. arctica, Hedysarum americanum, Hedysarum boreale subsp. mackenziei, Oxytropis hyperborea, Salix niphoclada, Salix reticulata, and Salix spp.

The map in Ball & Zoladz (1994) records this taxon’s eastern limit in western Nunavut, just beyond the territorial border, based on the only record they knew from the territory (Cox Lake, 60 miles W of Coppermine [Kugluktuk], 67.8833°N, 116.6333°W, August 1986, S. Fleck s.n., DAO-543874, det. P. Ball). It was subsequently recorded from the upper Hood River, as Carex petricosa (Gould & Walker, 1997); this locality is its current known eastern limit. Despite these previous Nunavut records, the taxon was not recorded for Nunavut in Ball & Mastrogiuseppe (2002). Bennett (2015) recently reported Carex petricosa from Ovayok (Mount Pelly) Territorial Park, near Cambridge Bay, Nunavut, in 2013, and from two additional sites in 2014, 30 Mile River and near Ferguson Lake and Ekalluk River on Victoria Island. The Mount Pelly collection (Bennett & Sullivan 13-0288, CAN-603847) was misidentified: the specimen is Carex fuliginosa subsp. misandra, and the others require confirmation. The nearest collections to the southwest are from the Great Bear Lake area (Porsild, 1943; Porsild & Cody, 1980), and the taxon occurs further north on northwestern Victoria Island (Aiken et al., 2007). It also reaches Arctic Canada in northern Yukon (Cody, 2000). The other infraspecific taxon, var. misandroides (Fernald) B. Boivin, treated at species level (Carex misandroides Fernald) in Porsild & Cody (1980), occurs in eastern Canada (Ball & Zoladz, 1994). We prefer to recognise these taxa as subspecies, since their ranges do not overlap and in accordance with a broader trend in the recent Arctic floristic literature to recognise infraspecific taxa as subspecies rather than varieties. We make the needed combination for the eastern taxon here:

Carex petricosa subsp. misandroides (Fernald) Saarela, comb. et stat. nov. Basionym: Carex misandroides Fernald, Rhodora 17: 158. 1915.

Specimens Examined: Canada. Nunavut: Kitikmeot Region: N side of Fockler Creek, ca. 1.9 km S of Sandstone Rapids, Coppermine River, 67°25′57.89″N, 115°38′3.9″W ± 10 m, 162 m, 4 July 2014, Saarela, Sokoloff & Bull 3318 (CAN, K, NY, QFA, UBC); forest and slopes at confluence of Big Creek and Coppermine River, N side of Coppermine River, S side of Coppermine Mountains, 67°14′29.3″N, 116°2′44.5″W ± 250 m, 180–199 m, 7 July 2014, Saarela, Sokoloff & Bull 3553 (ALTA, CAN, O); SE-facing slopes above Escape Rapids, W side of Coppermine River, 67°36′49.8″N, 115°29′27.4″W ± 10 m, 67 m, 8 July 2014, Saarela, Sokoloff & Bull 3736 (CAN, UBC); Kugluk (Bloody Falls) Territorial Park, top of sandy ridge, ca. 0.75 km W of Bloody Falls, 67°44′45.7″N, 115°23′4.6″W ± 25 m, 56 m, 15 July 2014, Saarela, Sokoloff & Bull 4042 (ALA, CAN); Kugluk (Bloody Falls) Territorial Park, W side of Coppermine River, between Sandy Hills and Bloody Falls, 67°45′13.2″N, 115°22′6.3″W ± 3 m, 21 m, 17 July 2014, Saarela, Sokoloff & Bull 4147 (CAN, MO, MT); Kugluk (Bloody Falls) Territorial Park, deep gully in sand hills, NW of Bloody Falls, 67°45′12.9″N, 115°22′54.1″W ± 3 m, 28 m, 18 July 2014, Saarela, Sokoloff & Bull 4173 (CAN, US, WIN).

Carex podocarpa R. Br., Fig. 25—Graceful mountain sedge | Asian (NE)–amphi-Beringian–Cordilleran | Noteworthy Record

Figure 25 Carex podocarpa.

(A) Inflorescence, Saarela et al. 3184. (B) Habit, Saarela et al. 4359. Photographs by R. D. Bull (A) and P. C. Sokoloff (B).

Previously recorded from Bloody Falls (Porsild & Cody, 1980), but we were unable to locate a voucher. We found this primarily boreal, tussock-forming taxon to be common in the study area, usually growing in meadows, low shrub tundra or snowbed communities. It occurs throughout the study area, as far north as Kugluktuk—a minor range northern range extension from the earlier Bloody Falls record—representing its known northern limit in Nunavut. We made numerous collections at Fockler Creek, Melville Creek, Coppermine Mountains, Kugluk (Bloody Falls) Territorial Park and Kugluktuk. It may be confused in the field with Carex atrofusca, but the latter is readily distinguished by its serrulate beak and upper perigynium margins (Murray, 1970). Recorded as far east as Bathurst Inlet, where known from a single collection (Kelsall & McEwen 271, CAN-202995; mapped in Porsild & Cody, 1980), and also recorded from the Hood River (Gould & Walker, 1997). There are no collections from between these areas and the study area. Elsewhere in Canada it reaches the Low Arctic west of the Mackenzie River Delta (Porsild, 1943) and in northern Yukon (Cody, 2000). Carex podocarpa was described by Robert Brown in Richardson (1823) based on material collected between Point Lake and the Arctic coast (i.e., along the Coppermine River), possibly from the study area (type BM000611688, GH00027375, K001079019).

Specimens Examined: Canada. Nunavut: Kitikmeot region: old riverbed of Fockler Creek, ca. 2.3 km SSE of Sandstone Rapids, Coppermine River, 67°25′45.7″N, 115°37′21.8″W ± 25 m, 166 m, 1 July 2014, Saarela, Sokoloff & Bull 3164 (ALTA, CAN, MO, O); old riverbed of Fockler Creek, ca. 2.3 km SSE of Sandstone Rapids, Coppermine River, 67°25′45.7″N, 115°37′21.8″W ± 25 m, 166 m, 2 July 2014, Saarela, Sokoloff & Bull 3184 (CAN, MT, US, WIN); NW-facing slope above tributary of Fockler Creek, ca. 2.4 km SSW of Sandstone Rapids, Coppermine River, 67°25′46″N, 115°38′49.4″W ± 50 m, 149 m, 3 July 2014, Saarela, Sokoloff & Bull 3291 (CAN, MO, O); NW-facing slope just upstream of small tributary from its confluence with Fockler Creek, ca. 2.4 km SSW of Sandstone Rapids, Coppermine River, 67°25′46″N, 115°38′49.4″W ± 200 m, 149 m, 3 July 2014, Saarela, Sokoloff & Bull 3304 (CAN, UBC); meadow just S of Tundra Lake, ca. 4.2 km SE of Sandstone Rapids, Coppermine River, 67°25′29.5″N, 115°33′50.4″W ± 50 m, 266 m, 5 July 2014, Saarela, Sokoloff & Bull 3431 (CAN, MT, US); confluence of Coppermine River and Melville Creek, just W of Coppermine Mountains, 67°15′52″N, 115°30′55.3″W ± 350 m, 178–190 m, 7 July 2014, Saarela, Sokoloff & Bull 3512 (ALA, CAN); S-facing slopes above Coppermine River and below spruce forest, ca. 7.8 km NNE of Sandstone Rapids, 67°31′16.2″N, 115°36′52.1″W ± 200 m, 110 m, 8 July 2014, Saarela, Sokoloff & Bull 3647 (CAN, QFA, WIN); flats atop and upper slopes of Coppermine Mountains, N/W side of Coppermine River, 67°14′53.6″N, 115°38′37.9″W ± 15 m, 401 m, 9 July 2014, Saarela, Sokoloff & Bull 3761 (CAN); Kugluk (Bloody Falls) Territorial Park, NE-facing slope of large hill just S of Bloody Falls, W side of Coppermine River, 67°44′6.6″N, 115°23′13.4″W ± 50 m, 40 m, 14 July 2014, Saarela, Sokoloff & Bull 3987 (CAN, K, NY); gradual slopes above small meadow that drain into a deep gully in sand hills NW of Bloody Falls, 67°45′40.2″N, 115°23′0.1″W ± 3 m, 75 m, 18 July 2014, Saarela, Sokoloff & Bull 4175 (ALTA, CAN); W of Kugluktuk on tundra flats above Coppermine River, S of 1 Coronation Drive and N of community power plant, 67°49′28.97″N, 115°5′0.2″W ± 100 m, 8 m, 22 July 2014, Saarela, Sokoloff & Bull 4268 (CAN); SE edge of Kugluktuk, rocky cliffs overlooking Coppermine River, 67°49′9.2″N, 115°5′40.4″W ± 50 m, 28 m, 24 July 2014, Saarela, Sokoloff & Bull 4359 (ALA, CAN, UBC).

Carex rariflora (Wahlenb.) Sm., Fig. S5A—Loose-flowered alpine sedge | Circumpolar

Previously recorded from Kugluktuk (Cody, 1954b; Porsild & Cody, 1980). We made collections at Fockler Creek, Kugluk (Bloody Falls) Territorial Park, and Kugluktuk. Infraspecific taxonomy is reviewed in Saarela et al. (2013a). In the Canadian Arctic recorded from Baffin, Coats, Southampton and Victoria islands, and other mainland sites (Porsild & Cody, 1980; Cody, Scotter & Zoltai, 1989; Korol, 1992; Cody & Reading, 2005; Aiken et al., 2007; Saarela et al., 2013a).

Specimens Examined: Canada. Nunavut: Kitikmeot region: Coppermine [Kugluktuk], 67°49′36″N, 115°5′36″W, 2 July 1951, W. I. Findlay 68 (DAO-175414 01-01000677834); S of Fockler Creek, along small tributary that runs into Fockler Creek, ca. 2.3 km S of Sandstone Rapids, Coppermine River, 67°25′44.9″N, 115°38′25.9″W ± 100 m, 152 m, 3 July 2014, Saarela, Sokoloff & Bull 3257 (ALA, CAN, UBC); Kugluk (Bloody Falls) Territorial Park, flats 1 km W of Bloody Falls, W side of Coppermine River, 67°44′31.5″N, 115°23′21.4″W ± 5 m, 68 m, 16 July 2014, Saarela, Sokoloff & Bull 4094 (ALTA, CAN, MO, MT, O); W of Kugluktuk on tundra flats above Coppermine River, S of 1 Coronation Drive and N of community power plant, 67°49′28.97″N, 115°5′0.2″W ± 100 m, 8 m, 22 July 2014, Saarela, Sokoloff & Bull 4269 (CAN, NY, QFA, US, WIN).

Carex rupestris All.—Rock sedge | Circumpolar-alpine

Previously recorded from Kugluktuk (Porsild & Cody, 1980). We made collections at Fockler Creek, Big Creek, Kugluk (Bloody Falls) Territorial Park, and Kugluktuk. In the Canadian Arctic this common dry tundra species is recorded from Baffin, Banks, Ellesmere, Melville, Somerset, Southampton and Victoria islands, and other mainland sites (Porsild & Cody, 1980; Korol, 1992; Aiken et al., 2007; Saarela et al., 2013a).

Specimens Examined: Canada. Nunavut: Kitikmeot Region: Coppermine [Kugluktuk], vicinity of post [67°49′36″N, 115°5′36″W ± 1.5 km], 23 July 1956, A. E. Porsild 17164 (CAN-127575); Kugluktuk, airport, 67.816667°N, 115.143889°W, 2 July 2008, L. J. Gillespie, J. M. Saarela, L. M. Consaul & R. D. Bull 7458 (CAN-592565); Kugluktuk, rocky slopes of North Hill, 67°49′31.4″N, 115°6′54″W ± 100 m, 42 m, 29 June 2014, Saarela, Sokoloff & Bull 3082 (CAN, MO, MT, O); second ridge N of Fockler Creek, ca. 1.9 km SSE of Sandstone Rapids, Coppermine River, 67°26′2.4″N, 115°37′26.5″W ± 25 m, 187 m, 2 July 2014, Saarela, Sokoloff & Bull 3213 (CAN, NY, QFA, US, WIN); forest and slopes at confluence of Big Creek and Coppermine River, N side of Coppermine River, S side of Coppermine Mountains, 67°14′29.3″N, 116°2′44.5″W ± 250 m, 180–199 m, 7 July 2014, Saarela, Sokoloff & Bull 3564 (CAN, UBC); Kugluk (Bloody Falls) Territorial Park, upper ledges of rocky (gabbro) S-facing cliffs above the start of Bloody Falls (W bank of River), just E of Portage Trail, 67°44′21.7″N, 115°22′42.2″W ± 25 m, 46 m, 14 July 2014, Saarela, Sokoloff & Bull 3939 (ALA, ALTA, CAN); W of Kugluktuk on tundra flats above Coppermine River, S of 1 Coronation Drive and N of power plant, 67°49′28.97″N, 115°5′0.2″W ± 100 m, 8 m, 25 July 2014, Saarela, Sokoloff & Bull 4375 (ALA, ALTA, CAN, K, MO, O, UBC).

Carex saxatilis L., Figs. S5B and S5C—Russet sedge | Circumboreal-polar

Previously recorded from Kugluktuk (Porsild & Cody, 1980). We made collections of this wetland sedge at Fockler Creek, Bigtree River, Kugluk (Bloody Falls) Territorial Park, and Kugluktuk. In the Canadian Arctic recorded from Baffin, Banks, Coats, Ellesmere, Southampton and Victoria islands, and other mainland sites (Porsild & Cody, 1980; Korol, 1992; Aiken et al., 2007; Saarela et al., 2013a). Taxonomic treatments of Carex saxatilis have varied (Ford & Ball, 1992; Elven et al., 2011). Porsild & Cody (1980) recognised plants in the study area as Carex saxatilis var. rhomalea Fernald, and also recognised the separate species Carex physocarpa Presl, which was treated as a synonym of Carex saxatilis by Ford & Ball (1992) and Reznicek & Ford (2002). We follow the latter treatments and recognise a polymorphic C. saxatilis in North America, whereas Elven et al. (2011) recognise North American plants as subsp. laxa (Trautv.) Kalela.

Specimens Examined: Canada. Nunavut: Kitikmeot region: Coppermine [Kugluktuk] [67°49′36″N, 115°5′36″W ± 1.5 km], 2 August 1962, J. A. Larsen s.n. (CAN-286317); S of Fockler Creek, along small tributary that runs into Fockler Creek, ca. 2.3 km S of Sandstone Rapids, Coppermine River, 67°25′44.9″N, 115°38′25.9″W ± 100 m, 152 m, 3 July 2014, Saarela, Sokoloff & Bull 3262 (ALTA, CAN, MO, O); confluence of Coppermine and Bigtree rivers, 66°56′23.8″N, 116°21′3.2″W ± 100 m, 265 m, 7 July 2014, Saarela, Sokoloff & Bull 3607 (CAN, MT, US, WIN); Kugluk (Bloody Falls) Territorial Park, rocky valley immediately SW of Bloody Falls, along rough marked section of Portage Trail, 67°44′34″N, 115°22′16″W ± 50 m, 20 m, 13 July 2014, Saarela, Sokoloff & Bull 3911 (CAN, K, NY, QFA, WIN); W of Kugluktuk on tundra flats above Coppermine River, S of 1 Coronation Drive and N of power plant, 67°49′28.97″N, 115°5′0.2″W ± 100 m, 8 m, 25 July 2014, Saarela, Sokoloff & Bull 4362 (ALA, CAN, UBC).

Carex scirpoidea Michx. subsp. scirpoidea, Fig. S6A—Scirpus sedge | Amphi-Beringian–North America (N)–amphi-Atlantic (W)

Previously recorded from Kugluktuk (Cody, 1954b; Porsild & Cody, 1980). We made collections at Fockler Creek, Melville Creek, Escape Rapids, and Kugluk (Bloody Falls) Territorial Park. In the Canadian Arctic recorded from Baffin, Banks, Coats, Devon, Ellesmere, King William, Southampton and Victoria islands, and other mainland sites (Porsild & Cody, 1980; Cody, Scotter & Zoltai, 1989; Korol, 1992; Cody & Reading, 2005; Aiken et al., 2007; Saarela et al., 2013a). Taxonomy follows recent treatments, which are in agreement with the recognition of infraspecific taxa (Dunlop & Crow, 1999; Dunlop, 2002; Elven et al., 2011).

Specimens Examined: Canada. Nunavut: Kitikmeot region: Coppermine [Kugluktuk], 67°49′36″N, 115°5′36″W, 23 June 1951, W. I. Findlay 31 (DAO-175100 01-01000677472); Coppermine [Kugluktuk] (67°49′36″N, 115°5′36″W ± 1.5 km), 2 July 1958, R. D. Wood s.n. (CAN-485074); flats on W side of Fockler Creek, above spruce forest in creek valley, ca. 2.2 km S of Sandstone Rapids, Coppermine River, 67°25′49″N, 115°37′55″W ± 50 m, 152 m, 1 July 2014, Saarela, Sokoloff & Bull 3128 (CAN, UBC); confluence of Coppermine River and Melville Creek, just W of Coppermine Mountains, 67°15′52″N, 115°30′55.3″W ± 350 m, 178–190 m, 7 July 2014, Saarela, Sokoloff & Bull 3504 (CAN, MO, MT, O, QFA, US, WIN); SE-facing slopes above Escape Rapids, W side of Coppermine River, 67°36′49.8″N, 115°29′27.4″W ± 10 m, 67 m, 8 July 2014, Saarela, Sokoloff & Bull 3740 (CAN); Kugluk (Bloody Falls) Territorial Park, rocky beach above Bloody Falls, W bank of Coppermine River, 67°44′18″N, 115°22′57.3″W ± 250 m, 34 m, 14 July 2014, Saarela, Sokoloff & Bull 3974 (ALA, ALTA, CAN).

Carex simpliciuscula subsp. subholarctica (T. V. Egorova) Saarela, comb. nov. Basionym: Kobresia simpliciuscula subsp. subholarctica T. V. Egorova, Novosti. Sist. Vyssh. Rast., 20:83. 1983—Simple bog sedge | Asian (NE)–amphi-Beringian–North American (N)–amphi-Atlantic (W)

Previously recorded from the study area (Porsild & Cody, 1980), but we were unable to locate a voucher specimen. We made collections at Fockler Creek, Melville Creek Big Creek, Coppermine Mountains and Kugluk (Bloody Falls) Territorial Park. Elsewhere in the Canadian Arctic recorded from Baffin, Banks, Devon, Ellesmere, Southampton and Victoria islands, and other mainland sites (Porsild & Cody, 1980; Cody, Scotter & Zoltai, 1989; Aiken et al., 2007; Saarela et al., 2013a). Previously recognised as Kobresia simpliciuscula subsp. subholarctica T. V. Egorova (Elven et al., 2011). Species of Kobresia are now recognised in Carex (Global Carex Group, 2015). The correct name for Kobresia simpliciuscula (Wahlenb.) Mack. in Carex is Carex simpliciuscula Wahlenb., and the combination for subsp. subholarctica is made here. The nominate subspecies is European and does not reach the Arctic (Elven et al., 2011).

Specimens Examined: Canada. Nunavut: Kitikmeot region: sedge meadow adjacent to small lake on flats N of Fockler Creek, ca. 1.5 km SSE of Sandstone Rapids, Coppermine River, 67°26′8.8″N, 115°37′35.9″W ± 20 m, 168 m, 2 July 2014, Saarela, Sokoloff & Bull 3224 (ALTA, CAN, O); confluence of Coppermine River and Melville Creek, just W of Coppermine Mountains, 67°15′52″N, 115°30′55.3″W ± 350 m, 178–190 m, 7 July 2014, Saarela, Sokoloff & Bull 3507 (CAN, MO, MT); forest and slopes at confluence of Big Creek and Coppermine River, N side of Coppermine River, S side of Coppermine Mountains, 67°14′29.3″N, 116°2′44.5″W ± 250 m, 180–199 m, 7 July 2014, Saarela, Sokoloff & Bull 3550 (CAN, UBC); Kugluk (Bloody Falls) Territorial Park, upper ledges of rocky (gabbro) S-facing cliffs above the start of Bloody Falls (W bank of River), just E of Portage Trail, 67°44′21.7″N, 115°22′42.2″W ± 25 m, 46 m, 14 July 2014, Saarela, Sokoloff & Bull 3942 (CAN, US, WIN); Kugluk (Bloody Falls) Territorial Park, W side of Coppermine River, between Sandy Hills and Bloody Falls, 67°45′13.2″N, 115°22′6.3″W ± 3 m, 21 m, 17 July 2014, Saarela, Sokoloff & Bull 4145 (ALA, CAN).

Carex subspathacea Wormsk., Figs. S6B and S6C—Hoppner’s sedge | Circumpolar | Noteworthy Record

Newly recorded for the study area. Our collections of this coastal and halophytic rhizomatous sedge from Richardson Bay and an island in the mouth of the Coppermine River close a distribution gap between Bernard Harbour to the northwest and Bathurst Inlet to the east (Macoun & Holm, 1921; Porsild & Cody, 1980). Elsewhere in the Canadian Arctic recorded from Baffin, Banks, Devon, Ellesmere, King William, Southampton and Victoria islands and a few other mainland sites (Porsild & Cody, 1980; Korol, 1992; Cody & Reading, 2005; Aiken et al., 2007; Saarela et al., 2013a).

Specimens Examined: Canada. Nunavut: Kitikmeot region: Richardson Bay, confluence of Richardson and Rae rivers at Coronation Gulf, ca. 20 km WNW of Kugluktuk, 67°54′11.2″N, 115°32′27.4″W ± 200 m, 0 m, 8 July 2014, Saarela, Sokoloff & Bull 3670 (CAN, UBC); unnamed island just E (ca. 3.3 km) of Kugluktuk at mouth of Coppermine River, 67°49′29.2″N, 115°1′3.2″W ± 50 m, 1 m, 8 July 2014, Saarela, Sokoloff & Bull 3724 (ALA, CAN).

Carex supina subsp. spaniocarpa (Steud.) Hultén—Weak Arctic sedge | Circumboreal-polar

Previously recorded from the study area (Porsild & Cody, 1980), but we were unable to locate a voucher specimen. We made collections of this rhizomatous sedge at Fockler Creek and Heart Lake, where it grew in disturbed areas. Elsewhere in the Canadian Arctic recorded from Baffin, Banks and Southampton islands, and a few other mainland sites (Porsild & Cody, 1980; Cody & Reading, 2005; Aiken et al., 2007).

Specimens Examined: Canada. Nunavut: Kitikmeot region: second ridge N of Fockler Creek, ca. 1.9 km SSE of Sandstone Rapids, Coppermine River, 67°26′2.4″N, 115°37′26.5″W ± 25 m, 187 m, 2 July 2014, Saarela, Sokoloff & Bull 3218 (CAN); disturbed slopes on N side of Fockler Creek, ca. 2 km SSE of Sandstone Rapids, 67°25′55.6″N, 115°37′42.7″W ± 9 m, 159 m, 2 July 2014, Saarela, Sokoloff & Bull 3744 (ALA, CAN); S of Fockler Creek, S-facing slope on N side of small tributary flowing into Fockler Creek, ca. 2.3 km S of Sandstone Rapids, Coppermine River, 67°25′46.3″N, 115°38′2.5″W ± 25 m, 156 m, 3 July 2014, Saarela, Sokoloff & Bull 3238 (CAN, UBC); E side of Fockler Creek, ridge above creek valley before its confluence with Coppermine River, ca. 1.8 km S of Sandstone Rapids, 67°26′3.9″N, 115°38′20.4″W ± 25 m, 168 m, 4 July 2014, Saarela, Sokoloff & Bull 3339 (CAN, QFA, US, WIN); SW-facing slope above (N side) of Fockler Creek, ca. 3.2 km SE of Sandstone Rapids, Coppermine River, 67°25′26.2″N, 115°36′14″W ± 25 m, 193 m, 5 July 2014, Saarela, Sokoloff & Bull 3414 (ALTA, CAN, O); Heart Lake, SW of Kugluktuk, 6.4 km SW of mouth of Coppermine River, 67°48′6.7″N, 115°13′40.6″W ± 50 m, 41 m, 23 July 2014, Saarela, Sokoloff & Bull 4305 (CAN, MO, MT).

Carex ursina Dewey—Bear sedge | Circumpolar | Noteworthy Record

Newly recorded for the study area. Our collection from Richardson Bay closes a distributional gap for this densely tufted seashore species between Bathurst Inlet and sites in the vicinity of Paulatuk (Porsild & Cody, 1980; Saarela et al., 2013a). Elsewhere in the Canadian Arctic recorded from Baffin, Banks, Coats, Devon, Ellesmere, King William, Melville, Prince Patrick, Somerset, Southampton and Victoria islands, and a few other mainland sites (Porsild & Cody, 1980; Korol, 1992; Aiken et al., 2007; Saarela et al., 2013a).

Specimens Examined: Canada. Nunavut: Kitikmeot region: Richardson Bay, confluence of Richardson and Rae rivers at Coronation Gulf, ca. 20 km WNW of Kugluktuk, 67°54′11.2″N, 115°32′27.4″W ± 200 m, 0 m, 8 July 2014, Saarela, Sokoloff & Bull 3675 (ALA, CAN, UBC).

Carex vaginata Tausch, Fig. S7—Sheathed sedge | Circumboreal-polar

Previously recorded from Kugluktuk (Cody, 1954b; Porsild & Cody, 1980). We made collections at Fockler Creek, Big Creek, Kugluk (Bloody Falls) Territorial Park, and Kugluktuk. Elsewhere in the Canadian Arctic recorded from southern Baffin Island, Banks, Southampton and Victoria islands, and a few other mainland sites (Porsild & Cody, 1980; Cody, Scotter & Zoltai, 1984, 1989; Korol, 1992; Cody & Reading, 2005; Aiken et al., 2007; Saarela et al., 2013a).

Specimens Examined: Canada. Nunavut: Kitikmeot region: Coppermine [Kugluktuk], 67°49′36″N, 115°5′36″W, W. I. Findlay 69 (DAO, not seen); Kugluktuk, rocky slopes of North Hill, 67°49′31.4″N, 115°6′54″W ± 100 m, 42 m, 29 June 2014, Saarela, Sokoloff & Bull 3070 (CAN); flats on W side of Fockler Creek, above spruce forest in creek valley, ca. 2.2 km S of Sandstone Rapids, Coppermine River, 67°25′49″N, 115°37′55″W ± 50 m, 152 m, 1 July 2014, Saarela, Sokoloff & Bull 3109 (ALA, ALTA, CAN, O); forest and slopes at confluence of Big Creek and Coppermine River, N side of Coppermine River, S side of Coppermine Mountains, 67°14′29.3″N, 116°2′44.5″W ± 250 m, 180–199 m, 7 July 2014, Saarela, Sokoloff & Bull 3566 (CAN, UBC); Kugluk (Bloody Falls) Territorial Park, flats above boardwalk W of Bloody Falls, 67°44′34.5″N, 115°22′27″W ± 100 m, 135 m, 13 July 2014, Saarela, Sokoloff & Bull 3919 (CAN, MO, MT, UBC).

Carex williamsii Britton—Williams’ sedge | Asian (N/C)–amphi-Beringian–North American (N) | Noteworthy Record

Our two collections are the first records of this uncommon species for the study area and fill in a distribution gap between Great Slave Lake, Bathurst Inlet, Hood River and the Tuktoyaktuk Peninsula and Eskimo Lakes area (Porsild, 1943; Porsild & Cody, 1980; Gould & Walker, 1997). The taxon was not recorded for Tuktut Nogait National Park and vicinity (Saarela et al., 2013a). At a site just south of Heart Lake it was locally common in wet, hummocky tundra with Andromeda polifolia, Betula glandulosa, Carex membranacea and Rubus chamaemorus. At a second site west of Kugluktuk it grew in an extremely lush sedge meadow along a small creek enriched by runoff from the nearby sewage retention pond, with Carex aquatilis subsp. stans, Descurainia sophioides, Eriophorum angustifolium, Tephroseris palustris subsp. congesta and Rubus chamaemorus. The study area is the northernmost limit for the species in Nunavut. Elsewhere in the Canadian Arctic known from southern Baffin Island, Southampton Island, adjacent northern Quebec and Labrador, and a few other sites on mainland Nunavut (Aiken et al., 2007).

Specimens Examined: Canada. Nunavut: Kitikmeot Region: ca. 0.5 km SW of Heart Lake, SW of Kugluktuk, 7.5 km SW of mouth of Coppermine River, 67°47′52″N, 115°14′14.4″W ± 350 m, 66 m, 23 July 2014, Saarela, Sokoloff & Bull 4292 (ALA, ALTA, CAN); creek just N of sewage retention pond (used as sewage outlet), 5.1 km SW of Coppermine River, 67°48′59.1″N, 115°12′5.8″W ± 25 m, 34 m, 23 July 2014, Saarela, Sokoloff & Bull 4333 (CAN, UBC).

Eleocharis acicularis (L.) Roem. & Schult.—Needle spikerush | Circumboreal-polar

Previously recorded from Kugluktuk (Cody, 1954b; Porsild & Cody, 1980), but we were unable to locate the voucher (Findlay 255B) for verification. We made collections of this rare species near the Kendall River, where it was locally common and submerged in shallow water, and at Heart Lake, where it formed dense patches on a wet sandy seasonally flooded beach, growing with Carex aquatilis subsp. stans, Juncus alpinoarticulatus subsp. americanus, Juncus arcticus subsp. alaskanus, Juncus triglumis subsp. albescens and Salix alaxensis. One of our collections (no. 3588) was sterile, while the other (no. 4299b) was gathered in flower. Elsewhere in the Canadian Arctic recorded from southern Baffin Island, Bathurst Inlet and southeastern mainland Nunavut (Porsild & Cody, 1980; Cody, Scotter & Zoltai, 1984, 1989; Aiken et al., 2007; Saarela et al., 2013a).

Specimens Examined: Canada. Nunavut: Kitikmeot region: confluence of Coppermine and Kendall rivers (NW side of Coppermine River, S side of Kendall River), small ponds on flats adjacent to Coppermine River, 67°6′44.7″N, 116°8′6.1″W ± 100 m, 213 m, 7 July 2014, Saarela, Sokoloff & Bull 3588 (CAN); Heart Lake, SW of Kugluktuk, 6.4 km SW of mouth of Coppermine River, 67°48′7.8″N, 115°13′22.7″W ± 350 m, 33 m, 23 July 2014, Saarela, Sokoloff & Bull 4299b (CAN).

Eleocharis quinqueflora (Hartmann) O. Schwarz, Fig. 26—Few-flowered spikerush | North American (NE) | Noteworthy Record

Figure 26 Eleocharis quinqueflora.

(A) Inflorescence, Saarela et al. 3971. (B) Habit, Saarela et al. 3971. Photographs by R. D. Bull.

Our two collections are the first records for Nunavut, and represent a northeastern range extension from the nearest known sites around Great Bear Lake (Porsild, 1943; Porsild & Cody, 1980). Both were made in the Arctic portion of the study area. The species was locally common in wet, muddy ground in a wet meadow near Kugluktuk, growing with Carex aquatilis subsp. stans, Carex membranacea, Eriophorum scheuchzeri, and Pinguicula vulgaris; and grew on a rocky beach just west of Bloody Falls with Artemisia tilesii, Bromus pumpellianus, Calamagrostis purpurascens, Carex saxatilis, Deschampsia cespitosa, Hedysarum americanum, Juncus arcticus subsp. alaskanus, Juncus leuchochlamys, Salix alaxensis and Salix niphoclada. Treatments of the taxon have varied. Porsild & Cody (1980) treated it as Eleocharis pauciflora var. fernaldii Svenson, Elven et al. (2011) treated it as Eleocharis quinqueflora subsp. fernaldii (Svenson) Hultén, whereas Smith et al. (2002) did not recognise infraspecific taxa, citing the need for further study. We follow the latter treatment. The key below, adapted from Smith et al. (2002), includes characters that may be used to separate this species from Eleocharis acicularis in the absence of mature achenes (used in the key in Porsild & Cody, 1980). Two of our collections (nos. 3971 and 4270) were in flower, but did not have mature achenes.

Key distinguishing Eleocharis acicularis and Eleocharis quinqueflora, adapted from Smith et al. (2002): 1. Anthers 1.5–2.7(–3.5) mm; achenes 1.6–2.3 × 0.7–1.3 mm, tubercles 0.3–0.4 × 0.2–0.3 mm, rarely absent; floral scales 2.5–6 × 1.5–2.5 mmEleocharis quinqueflora

1′ Anthers 0.7–1.5 mm; achenes 0.7–1.1 × 0.35–0.6 mm, tubercles (0.05–) 0.1–0.2 × 0.15–0.25 mm; floral scales 1.5–2.5 (–3.5) × 1–1.5 mmEleocharis acicularis

Specimens Examined: Canada. Nunavut: Kitikmeot region: Kugluk (Bloody Falls) Territorial Park, rocky beach above Bloody Falls, W bank of Coppermine River, 67°44′18″N, 115°22′57.3″W ± 250 m, 34 m, 14 July 2014, Saarela, Sokoloff & Bull 3971 (CAN, UBC); W of Kugluktuk on tundra flats above Coppermine River, S of 1 Coronation Drive and N of community power plant, 67°49′28.97″N, 115°5′0.2″W ± 100 m, 8 m, 22 July 2014, Saarela, Sokoloff & Bull 4270 (ALA, ALTA, CAN).

Eriophorum angustifolium Honck., Figs. S8A and S8B—Narrow-leaved cottongrass | Circumboreal-polar

Previously recorded from Kugluktuk (Cody, 1954b; Porsild & Cody, 1980). We made collections at Fockler Creek and Kugluk (Bloody Falls) Territorial Park. Elsewhere in the Canadian Arctic recorded from Baffin, Banks, Coats, Ellesmere, Melville, Prince of Wales, Somerset, Southampton and Victoria islands, and sites across the mainland (Porsild & Cody, 1980; Cody, Scotter & Zoltai, 1989; Cody & Reading, 2005; Aiken et al., 2007; Saarela et al., 2013a). This taxon and Eriophorum triste have been recognised as separate species (Porsild & Cody, 1980; Elven et al., 2011), subspecies of Eriophorum angustifolium (Ball & Wujek, 2002) or a single polymorphic taxon (Aiken et al., 2007). We agree with Elven et al. (2011) in recognising them as species; a key to distinguish them is given in Saarela et al. (2013a). This species grows in slightly wetter habitats than Eriophorum triste.

Specimens Examined: Canada. Nunavut: Kitikmeot Region: Coppermine River, Fort Hearne [Kugluktuk] [67°49′36″N, 115°5′36″W ± 1.5 km], 1931, A. M. Berry 2 (CAN-28008); Coppermine [Kugluktuk], 67°49′36″N, 115°5′36″W, 3 August 1951, W. I. Findlay 245 (ACAD-30942, ALTA-VP-4195, DAO-174448 01-01000677911, UBC-V40776); Coppermine [Kugluktuk], 67°49′36″N, 115°5′36″W, 30 June 1951, W. I. Findlay 54 (DAO-174471 01-01000677912); Coppermine [Kugluktuk], 67°51′N, 115°16′W, 2 July 1972, F. Fodor N 145 (UBC-V151908); NW-facing slope above tributary of Fockler Creek, ca. 2.4 km SSW of Sandstone Rapids, Coppermine River, 67°25′46″N, 115°38′49.4″W ± 50 m, 149 m, 3 July 2014, Saarela, Sokoloff & Bull 3301 (ALA, ALTA, CAN); Sleigh Creek, near its confluence with small, unnamed tributary, ca. 1 km SE of Sandstone Rapids, Coppermine River, 67°27′2″N, 115°37′28.7″W ± 10 m, 150 m, 6 July 2014, Saarela, Sokoloff & Bull 3465 (CAN, MO, O); Kugluk (Bloody Falls) Territorial Park, flats on top of mountain on W side of Coppermine River, just S of the start of Bloody Falls Rapids, 67°43′58″N, 115°24′33.3″W ± 25 m, 109 m, 14 July 2014, Saarela, Sokoloff & Bull 4003 (CAN, UBC).

Eriophorum brachyantherum Trautv. & C. A. Mey.—Closed-sheath cottongrass | Circumboreal-polar

Previously recorded from Kugluktuk (Porsild & Cody, 1980). We made collections at Fockler Creek and just beyond the southwestern boundary of Kugluk (Bloody Falls) Territorial Park. The range of this species was recently expanded to northwestern Victoria Island, the first records for the western Canadian Arctic Archipelago (Gillespie et al., 2015). Elsewhere in the Canadian Arctic recorded from Baffin and Southampton islands, and a few other mainland sites (Porsild & Cody, 1980; Korol, 1992; Aiken et al., 2007; Saarela et al., 2013a).

Specimens Examined: Canada. Nunavut: Kitikmeot Region: Coppermine [Kugluktuk], vicinity of post [67°49′36″N, 115°5′36″W ± 1.5 km], 26 July 1949, A. E. Porsild 17165 (CAN-127517); sedge meadow adjacent to small lake on flats N of Fockler Creek, ca. 1.5 km SSE of Sandstone Rapids, Coppermine River, 67°26′8.8″N, 115°37′35.9″W ± 20 m, 168 m, 2 July 2014, Saarela, Sokoloff & Bull 3228 (CAN, MT, US); Sleigh Creek, near its confluence with small, unnamed tributary, ca. 1 km SE of Sandstone Rapids, Coppermine River, 67°27′2″N, 115°37′28.7″W ± 10 m, 150 m, 6 July 2014, Saarela, Sokoloff & Bull 3469 (CAN, K, NY, UBC); flats on top of mountain on W side of Coppermine River, just S of the start of Bloody Falls Rapids, 67°43′52.3″N, 115°24′36.4″W ± 3 m, 115 m, 14 July 2014, Saarela, Sokoloff & Bull 4000 (CAN, QFA, WIN).

Eriophorum callitrix Cham.—Arctic cottongrass | Asian (N)–amphi-Beringian–North American (N) | Noteworthy Record

Newly recorded for the study area, closing a distribution gap between Bathurst Inlet, Hood River, Great Slave Lake, Tuktut Nogait National Park and vicinity, and sites on Victoria Island (Porsild & Cody, 1980; Gould & Walker, 1997; Saarela et al., 2013a). We made collections at Fockler Creek and Kugluk (Bloody Falls) Territorial Park. Elsewhere in the Canadian Arctic recorded from Baffin, Banks, Devon, Somerset, Southampton and Victoria islands, and other mainland sites (Porsild & Cody, 1980; Cody, Scotter & Zoltai, 1989; Korol, 1992; Aiken et al., 2007; Saarela et al., 2013a).

Specimens Examined: Canada. Nunavut: Kitikmeot Region: S of Fockler Creek, along small tributary that runs into Fockler Creek, ca. 2.3 km S of Sandstone Rapids, Coppermine River, 67°25′44.9″N, 115°38′25.9″W ± 100 m, 152 m, 3 July 2014, Saarela, Sokoloff & Bull 3255 (CAN); E side of Fockler Creek, in valley just above creek’s confluence with the Coppermine River, ca. 1.4 km SSW of Sandstone Rapids, 67°26′14.5″N, 115°38′34.8″W ± 50 m, 146 m, 4 July 2014, Saarela, Sokoloff & Bull 3354 (CAN, UBC); Kugluk (Bloody Falls) Territorial Park, wet meadow between Coppermine River and large sand hills on W side of river, 0.5 km W of Bloody Falls, 67°44′44.8″N, 115°22′48.3″W ± 15 m, 33 m, 15 July 2014, Saarela, Sokoloff & Bull 4046 (CAN).

Eriophorum scheuchzeri subsp. arcticum M. S. Novos., Fig. S8C—Scheuchzer’s cottongrass | Circumpolar

Previously recorded from Kugluktuk, as Eriophorum scheuchzeri Hoppe (Cody, 1954b; Porsild & Cody, 1980). We made collections of this rhizomatous cottongrass at Fockler Creek, Escape Rapids, Kugluk (Bloody Falls) Territorial Park, Heart Lake and Kugluktuk. Taxonomy follows Elven et al. (2011), who recognise two subspecies in Eriophorum scheuchzeri (also see Cayouette, 2004). The taxonomic history of this complex is reviewed in Saarela et al. (2013a). Elsewhere in the Canadian Arctic recorded from Baffin, Banks, Coast, Devon, Ellesmere, Melville, Nottingham, Somerset, Southampton and Victoria islands (Aiken et al., 2007). Its mainland distribution is unclear, but this subspecies is known from Tuktut Nogait National Park and vicinity (Saarela et al., 2013a).

Specimens Examined: Canada. Nunavut: Kitikmeot Region: Coppermine [Kugluktuk], 67°49′36″N, 115°5′36″W, 6 August 1951, W. I. Findlay 256 (ACAD-30940, ALTA-VP-4239, DAO-174576 01-01000677914, MT00185938, UBC-V40774); Coppermine [Kugluktuk], 13 July 1958, R. D. Wood s.n. (CAN-265598); Coppermine [Kugluktuk], 67°51′N, 115°16′W, 2 July 1972, F. Fodor N 144 (UBC-V151909); Kugluktuk, 12 July 2006, 67 49.531 N, 115 05.415′W, J. D. Davis 625 (CAN-597654); E side of Fockler Creek, just above its confluence with Coppermine River, ca. 1.1 km SW of Sandstone Rapids, 67°26′30.6″N, 115°39′4.3″W ± 50 m, 135 m, 4 July 2014, Saarela, Sokoloff & Bull 3367 (CAN, UBC); confluence of Sleigh Creek and Coppermine River, 0.4 km N of Sandstone Rapids, 67°27′13.9″N, 115°38′7.7″W ± 25 m, 126 m, 6 July 2014, Saarela, Sokoloff & Bull 3482 (ALA, CAN); SE-facing slopes above Escape Rapids, W side of Coppermine River, 67°36′49.8″N, 115°29′27.4″W ± 10 m, 67 m, 8 July 2014, Saarela, Sokoloff & Bull 3743 (CAN, US, WIN); Kugluk (Bloody Falls) Territorial Park, rocky valley immediately SW of Bloody Falls, along rough marked section of Portage Trail, head of small unnamed pond just W of falls, 67°44′42.8″N, 115°22′29.2″W ± 10 m, 9 m, 13 July 2014, Saarela, Sokoloff & Bull 3907 (ALTA, CAN); Kugluk (Bloody Falls) Territorial Park, flats on top of mountain on W side of Coppermine River, just S of the start of Bloody Falls Rapids, 67°43′58″N, 115°24′33.3″W ± 25 m, 109 m, 14 July 2014, Saarela, Sokoloff & Bull 4004 (CAN, O); Kugluk (Bloody Falls) Territorial Park, W side of Coppermine River, just above Bloody Falls, 67°44′22.6″N, 115°22′52″W ± 20 m, 40m, 16 July 2014, Saarela, Sokoloff & Bull 4111 (CAN, MO); W of Kugluktuk on tundra flats above Coppermine River, S of 1 Coronation Drive and N of community power plant, 67°49′28.97″N, 115°5′0.2″W ± 100 m, 8 m, 22 July 2014, Saarela, Sokoloff & Bull 4264 (CAN, UBC); Heart Lake, SW of Kugluktuk, 6.4 km SW of mouth of Coppermine River, 67°48′7.8″N, 115°13′22.7″W ± 350 m, 33 m, 23 July 2014, Saarela, Sokoloff & Bull 4299a (CAN, K, NY, QFA).

Eriophorum triste (Th. Fr.) Hadač & Á. Löve, Figs. 27A and 27B—Tall cottongrass | Amphi-Beringian (E)—North American (N)—amphi-Atlantic (W) | Noteworthy Record

Figure 27 Eriophorum triste and Eriophorum vaginatum subsp. vaginatum.

Eriophorum triste: (A) habitat, Saarela et al. 4087. (B) Inflorescence, Saarela et al. 4087. Eriophorum vaginatum subsp. vaginatum: (C) habitat, Saarela et al. 4055. Photographs by P. C. Sokoloff (A, B) and J. M. Saarela (C).

Newly recorded for the study area. Our collections, from Fockler Creek and Kugluk (Bloody Falls) Territorial Park, close a distribution gap between Bernard Harbour to the northwest, eastern Great Bear Lake and central Nunavut (Porsild & Cody, 1980). See additional comments under Eriophorum angustifolium. Widespread across the Canadian Arctic Archipelago and known from a few other mainland sites (Porsild & Cody, 1980; Cody & Reading, 2005; Aiken et al., 2007; Saarela et al., 2013a).

Specimens Examined: Canada. Nunavut: Kitikmeot Region: E side of Fockler Creek, in valley just above creek’s confluence with the Coppermine River, ca. 1.4 km SSW of Sandstone Rapids, 67°26′14.5″N, 115°38′34.8″W ± 50 m, 146 m, 4 July 2014, Saarela, Sokoloff & Bull 3353 (ALA, CAN); Kugluk (Bloody Falls) Territorial Park, flats on top of mountain on W side of Coppermine River, just S of the start of Bloody Falls Rapids, 67°44′2.8″N, 115°23′39.3″W ± 250 m, 110 m, 14 July 2014, Saarela, Sokoloff & Bull 3990 (ALTA, CAN); Kugluk (Bloody Falls) Territorial Park, flat terrace just above S-facing cliffs above start of Bloody Falls, W side of Coppermine River, W side of Portage Trail, 67°44′41.3″N, 115°22′53.5″W ± 3 m, 65 m, 16 July 2014, Saarela, Sokoloff & Bull 4087 (CAN, MO, MT, O).

Eriophorum vaginatum L. subsp. vaginatum, Fig. 27C—Sheathed cottongrass | European–Asian–amphi-Beringian–North American (NW)

Previously recorded from Kugluktuk (Cody, 1954b; Porsild & Cody, 1980). We were unable to locate the voucher for one of the earlier records (Findlay 30) for verification. We made collections at Fockler Creek and Kugluk (Bloody Falls) Territorial Park. Elsewhere in the Canadian Arctic recorded from Banks, Melville and Victoria islands, and other mainland sites (Porsild & Cody, 1980; Cody, Scotter & Zoltai, 1984; Cody & Reading, 2005; Aiken et al., 2007; Saarela et al., 2013a). Although they did not recognise them as such, records for Baffin Island (at least the southern ones) in Aiken et al. (2007) likely correspond to subsp. spissum (Fernald) Hultén, an eastern taxon in the Canadian Arctic.

Specimens Examined: Canada. Nunavut: Kitikmeot Region: Coppermine [Kugluktuk], 67°49′36″N, 115°5′36″W, 4 June 1951, W. I. Findlay 2 (ACAD-30938, ALTA-VP-4270, DAO-174632 01-01000677851, UBC-V40772); Coppermine [Kugluktuk], 67°49′36″N, 115°5′36″W, 6 August 1951, W. I. Findlay 259 (ALTA-VP-4269, DAO-174631 01-01000677397); NW-facing slope just upstream of small tributary from its confluence with Fockler Creek, ca. 2.4 km SSW of Sandstone Rapids, Coppermine River, 67°25′46″N, 115°38′49.4″W ± 200 m, 149 m, 3 July 2014, Saarela, Sokoloff & Bull 3305 (CAN, UBC); Kugluk (Bloody Falls) Territorial Park, flats above boardwalk W of Bloody Falls, 67°44′34.5″N, 115°22′27″W ± 100 m, 135 m, 16 July 2014, Saarela, Sokoloff & Bull 4055 (ALA, ALTA, CAN).

Trichophorum cespitosum (L.) Hartm. subsp. cespitosum—Tufted bulrush | Circumboreal-polar

Previously recorded from the study area (Porsild & Cody, 1980), as Scirpus caespitosus subsp. austriacus, but we were unable to locate a voucher specimen. Elsewhere in the Canadian Arctic recorded from southern Baffin Island and some mainland sites (Porsild & Cody, 1980; Cody, Scotter & Zoltai, 1989; Korol, 1992; Aiken et al., 2007; Saarela et al., 2013a). The other subspecies is Atlantic European and does not reach the Arctic (Elven et al., 2011).

Specimens Examined: Canada. Nunavut: Kitikmeot Region: Kugluktuk, airport, 67.816667°N, 115.143889°W, 2 July 2008, L. J. Gillespie, J. M. Saarela, L. M. Consaul & R. D. Bull 7459 (CAN-592566); flats on W side of Fockler Creek, above spruce forest in creek valley, ca. 2.2 km S of Sandstone Rapids, Coppermine River, 67°25′49″N, 115°37′55″W ± 50 m, 152 m, 1 July 2014, Saarela, Sokoloff & Bull 3127 (ALA, CAN, UBC); Kugluk (Bloody Falls) Territorial Park, upper ledges of rocky (gabbro) S-facing cliffs above the start of Bloody Falls (W bank of River), just E of Portage Trail, 67°44′21.7″N, 115°22′42.2″W ± 25 m, 46 m, 14 July 2014, Saarela, Sokoloff & Bull 3941 (ALTA, CAN, O).

Juncaceae [2/8]

Juncus alpinoarticulatus subsp. americanus (Farwell) Hämet-Ahti, Figs. 28A and 28C—Alpine rush | Amphi-Beringian–North American (N) | Noteworthy Record

Figure 28 Juncus alpinoarticulatus subsp. americanus and Juncus arcticus subsp. alaskanus.

Juncus alpinoarticulatus subsp. americanus: (A) habit, Saarela et al. 3858. (C) habitat, Saarela et al. 3858. Juncus arcticus subsp. alaskanus: (B) inflorescence, Saarela et al. 3720. Photographs by J. M. Saarela (A, C) and P. C. Sokoloff (B).

Newly recorded for mainland Nunavut. We made three collections in Kugluk (Bloody Falls) Territorial Park, where the species grew on sandy, rocky beaches, mud flats and wet meadows, variously associated with Artemisia tilesii, Chamerion latifolium, Equisetum arvense, Juncus arcticus, Juncus leucochlamys and Juncus triglumis subsp. albescens. We made a fourth collection further north along the south shore of Heart Lake, where the species grew on a sandy, seasonally flooded beach with Carex aquatilis subsp. stans, Eriophorum spp., Juncus arcticus subsp. alaskanus and Juncus triglumis subsp. albescens. Elsewhere in Nunavut recorded from Akimiski Island (Blaney & Kotanen, 2001) and an island in Long Island Sound along the east coast of Hudson Bay (Hustich s.n., CAN-204979). It is not recorded for Nunavut in most other publications (Porsild & Cody, 1980; Hämet-Ahti, 1986; Brooks & Clemants, 2002; Kirschner, 2002a).

Juncus alpinoarticulatus is the correct name for plants that have been recognised as Juncus alpinus Vill., nom. illeg., as in Porsild & Cody (1980) (Hämet-Ahti, 1980b). This northern temperate to boreal taxon is the common subspecies in North America, ranging from Greenland and northeastern North America to Colorado and Alaska, and to Kamchatka and Chukotka (Hämet-Ahti, 1986). The other North American taxon, Juncus alpinoarticulatus subsp. fuscescens (Fernald) Hämet-Ahti, has a restricted distribution from New York to Minnesota (Hämet-Ahti, 1986), and four other subspecies are recognised in Eurasia (Hämet-Ahti, 1980a). Brooks & Clemants (2002) do not recognise infraspecific taxa. We accept the taxonomy proposed by Hämet-Ahti (1986), at least for northern plants, as followed by Elven et al. (2011).

Subspecies americanus is a variable taxon with respect to plant size and the form and colour of the inflorescences; plants with dark flowers and short peduncles are more common in the western and northern parts of the North American range (Hämet-Ahti, 1986). Our collections are at the low end of this variation, ranging from 7 to 14 cm in height, with (1–)2–3(–4) heads, outside the low end of the range (“inflorescences of 5–25 heads”) reported in Brooks & Clemants (2002), and partly inside the range of “(3–)10–20(–40) heads” reported by Hämet-Ahti (1986) and Kirschner (2002a). There are a few northern collections comprising plants with a similar short stature, few heads and short peduncles, from Great Bear Lake (N shore of Smith Arm, Olmsted Bay, ca. 66°32′N, 122°35′W, Porsild & Porsild 5075, CAN-11365, det. Hämet-Ahti, 1986; head of Hornby Bay, NE end of McTavish Arm, 66°28′N, 118°05′W, Shacklette 3170, CAN-199937, det. Hämet-Ahti, 1986), Aubry Lake (67°20′N, 126°25′W, Riewe & J. Marsh 425, CAN-434768, det. Hämet-Ahti, 1986). These previous collections and our new ones are all from the northern limits of the taxon’s range, and their reduced statures may simply reflect the harsh Subarctic to Low Arctic environments in which they grew.

Specimens Examined: Canada. Nunavut: Kitikmeot Region: Kugluk (Bloody Falls) Territorial Park, rocky valley immediately SW of Bloody Falls, along rough marked section of Portage Trail, 67°44′34″N, 115°22′16″W ± 50 m, 20 m, 13 July 2014, Saarela, Sokoloff & Bull 3858 (CAN, UBC); Kugluk (Bloody Falls) Territorial Park, rocky beach above Bloody Falls, W bank of Coppermine River, 67°44′18″N, 115°22′57.3″W ± 250 m, 34 m, 14 July 2014, Saarela, Sokoloff & Bull 3969 (ALA, CAN); Kugluk (Bloody Falls) Territorial Park, rocky sandy beach just below Bloody Falls, W side of Coppermine River, vicinity of confluence with small creek, beach seasonally flooded, 67°44′54.5″N, 115°22′17.2″W ± 75 m, 9 m, 17 July 2014, Saarela, Sokoloff & Bull 4118 (CAN); Heart Lake, SW of Kugluktuk, 6.4 km SW of mouth of Coppermine River, 67°48′7.8″N, 115°13′22.7″W ± 350 m, 33 m, 23 July 2014, Saarela, Sokoloff & Bull 4294 (ALTA, CAN, O).

Juncus arcticus subsp. alaskanus Hultén, Fig. 28B—Arctic bog rush | Circumpolar-alpine | Noteworthy Record

Collections made in 1999 and 2013 in Kugluktuk, and our new collections from Fockler Creek, Kugluk (Bloody Falls) Territorial Park and an island at the mouth of the Coppermine River, are the first records for the study area. The taxon was recently recorded at sites outside the study area: the Big Bend area of the Coppermine River (Reading 35-1, DAO; Cody, Reading & Line, 2003) and east of the confluence of the Coppermine and Kendall rivers (Reading 15B, DAO; Cody & Reading, 2005). All these collections close a distribution gap between Bernard Harbour, eastern Great Slave Lake, Bathurst Inlet and Hood River (Porsild & Cody, 1980; Gould & Walker, 1997). This western subspecies extends as far east as Bathurst Inlet and southeastern mainland Nunavut (Porsild & Cody, 1980; Kirschner, 2002b; Cody, Reading & Line, 2003), and to the north is recorded from Banks and Victoria islands (Aiken et al., 2007).

The name Juncus balticus subsp. alaskanus (Hultén) A. E. Porsild, comb. illeg., as used by Porsild & Cody (1980), is a synonym of Juncus arcticus subsp. alaskanus, and the plants treated as boreal-distributed Juncus balticus var. littoralis Engelm. in Porsild & Cody (1980) are now recognised as Juncus balticus subsp. ater (Rydb.) Snogerup (a western race) and Juncus balticus subsp. littoralis (Engelm.) Snogerup (Kirschner, 2002b; Snogerup, Zika & Kirschner, 2002). Brooks & Clemants (2002) recognised three varieties of Juncus arcticus Willd. for North America: var. arcticus, var. alaskanus (Hultén) S. L. Welsh and var. balticus (Willd.) Trautv. Elven et al. (2011), following Kirschner (2002a) as we do here, recognised Juncus balticus as a predominantly non-Arctic taxon, and two subspecies of Juncus arcticus in North America: subsp. arcticus (Greenland and northeastern Canada) and subsp. alaskanus Hultén (northwestern Canada and Alaska).

Specimens Examined: Canada. Nunavut: Kitikmeot Region: Coppermine [Kugluktuk], vic. of hamlet and airstrip, 67.78°N, 115.5°W ± 3,615 m, 23 June 1999, C. L. Parker & I. Jonsdottir 9105 (ALA, as Juncus arcticus); Kugluktuk, airport, 21 July 2013, 67.81749°N, 115.13449°W, B. A. Bennett 13-0333 (BABY, det. Juncus arcticus, B. A. Bennett, July 2013); E side of Fockler Creek, just above its confluence with Coppermine River, ca. 1.1 km SW of Sandstone Rapids, 67°26′30.6″N, 115°39′4.3″W ± 50 m, 135 m, 4 July 2014, Saarela, Sokoloff & Bull 3365 (CAN, QFA, US, WIN); unnamed island just E (ca. 3.3 km) of Kugluktuk at mouth of Coppermine River, 67°49′29.2″N, 115°1′3.2″W ± 50 m, 1 m, 8 July 2014, Saarela, Sokoloff & Bull 3720 (CAN); Kugluk (Bloody Falls) Territorial Park, rocky cliffs and ledges directly above (W side) of Bloody Falls, just S of heavily used day-use/fishing area, 67°44′40.1″N, 115°22′4.9″W ± 20 m, 8 m, 12 July 2014, Saarela, Sokoloff & Bull 3811 (CAN, MO); Kugluk (Bloody Falls) Territorial Park, terrace above S-facing slopes above start of Bloody Falls, W side of Coppermine River, 67°44′27.2″N, 115°22′58″W ± 50 m, 68 m, 16 July 2014, Saarela, Sokoloff & Bull 4091 (CAN, MT).

Juncus biglumis L.—Two-flowered bog rush | Circumpolar-alpine

Previously recorded from the study area (Porsild & Cody, 1980), but we were unable to locate a voucher specimen. This is a rare species in the study area. We encountered it at one site in Kugluk (Bloody Falls) Territorial Park, where it was scattered and uncommon in small mud boils in mesic low shrub tundra with Betula glandulosa, Carex bigelowii, Eriophorum vaginatum subsp. vaginatum, Rhododendron lapponicum and Rumex arcticus. Widespread throughout the Canadian Arctic Archipelago and known from numerous mainland sites (Porsild & Cody, 1980; Cody, Scotter & Zoltai, 1989; Korol, 1992; Aiken et al., 2007; Hay & Payette, 2013; Saarela et al., 2013a).

Specimens Examined: Canada. Nunavut: Kitikmeot Region: Kugluk (Bloody Falls) Territorial Park, flats above boardwalk W of Bloody Falls, 67°44′34.5″N, 115°22′27″W ± 100 m, 135 m, 18 July 2014, Saarela, Sokoloff & Bull 4176 (CAN, UBC).

Juncus leucochlamys V. J. Zinger ex V. I. Krecz., Fig. S9—Chestnut rush | Asian (N/C)–amphi-Beringian–North America (N)–amphi-Atlantic (W)

Previously recorded from the study area, as Juncus castaneus Smith (Porsild & Cody, 1980), but we were unable to locate a voucher specimen. We made collections at Fockler Creek and Kugluk (Bloody Falls) Territorial Park. In the Canadian Arctic recorded from Baffin, Devon, Ellesmere, Southampton and Victoria islands, and numerous mainland sites (Porsild & Cody, 1980; Cody, Scotter & Zoltai, 1989; Korol, 1992; Cody & Reading, 2005; Aiken et al., 2007; Hay & Payette, 2013; Saarela et al., 2013a). Taxonomy follows Elven, Murray & Solstad (2010) and Elven et al. (2011), who recognise Juncus leuchochlamys and Juncus castaneus as distinct species, with only the former present in North America. This treatment contrasts with that of Kirschner (2002a), who recognised infraspecific taxa (Juncus castaneus subsp. castaneus and subsp. leucochlamys (V. J. Zinger ex V. I. Krecz.) Hultén) as we did previously (Saarela et al., 2013a), and Brooks & Clemants (2002), who did not recognise infraspecific taxa in Juncus castaneus s.l.

Specimens Examined: Canada. Nunavut: Kitikmeot Region: S of Fockler Creek, along small tributary that runs into Fockler Creek, ca. 2.3 km S of Sandstone Rapids, Coppermine River, 67°25′44.9″N, 115°38′25.9″W ± 100 m, 152 m, 3 July 2014, Saarela, Sokoloff & Bull 3270 (CAN); gravel bar in Fockler Creek just above its confluence with Coppermine River, ca. 1.4 km SSW of Sandstone Rapids, 67°26′21.4″N, 115°38′54″W ± 5 m, 141 m, 4 July 2014, Saarela, Sokoloff & Bull 3364 (ALA, CAN); Kugluk (Bloody Falls) Territorial Park, rocky valley immediately SW of Bloody Falls, along rough marked section of Portage Trail, 67°44′34″N, 115°22′16″W ± 50 m, 20 m, 13 July 2014, Saarela, Sokoloff & Bull 3854 (ALTA, CAN).

Juncus triglumis subsp. albescens (Lange) Hultén—Northern white rush | Asian (N)–amphi-Beringian–North American (N)–amphi-Atlantic (W) | Circumpolar-alpine

Previously recorded from the study area, as Juncus albescens Lange (Porsild & Cody, 1980), but we were unable to locate a voucher specimen. We made collections at Fockler Creek and in Kugluk (Bloody Falls) Territorial Park. In the Canadian Arctic recorded from Baffin, Banks, Coats, Devon, Ellesmere, King William, Southampton and Victoria islands, and several mainland sites (Porsild & Cody, 1980; Cody, Scotter & Zoltai, 1989; Korol, 1992; Cody & Reading, 2005; Aiken et al., 2007; Hay & Payette, 2013; Saarela et al., 2013a). Elven et al. (2011) also recognised the taxon at species level, while Brooks & Clemants (2002) recognised it as var. albescens Lange.

Specimens Examined: Canada. Nunavut: Kitikmeot Region: sedge meadow adjacent to small lake on flats N of Fockler Creek, ca. 1.5 km SSE of Sandstone Rapids, Coppermine River, 67°26′8.8″N, 115°37′35.9″W ± 20 m, 168 m, 2 July 2014, Saarela, Sokoloff & Bull 3229 (CAN); slopes on E side of Coppermine River, N of its confluence with Fockler Creek, ca. 0.8 km SW of Sandstone Rapids, 67°26′36.9″N, 115°38′50.1″W ± 50 m, 128 m, 4 July 2014, Saarela, Sokoloff & Bull 3396 (CAN, UBC); Kugluk (Bloody Falls) Territorial Park, rocky valley immediately SW of Bloody Falls, along rough marked section of Portage Trail, 67°44′34″N, 115°22′16″W ± 50 m, 20 m, 13 July 2014, Saarela, Sokoloff & Bull 3855 (ALA, CAN).

Luzula confusa Lindeb.—Northern wood rush | Circumpolar-alpine

Previously recorded from Kugluktuk (Porsild & Cody, 1980). We made collections at Fockler Creek, Coppermine Mountains, Kugluk (Bloody Falls) Territorial Park, Heart Lake and Kugluktuk. Widespread throughout the Canadian Arctic (Porsild & Cody, 1980; Cody, Scotter & Zoltai, 1989; Korol, 1992; Aiken et al., 2007; Hay & Payette, 2013; Saarela et al., 2013a).

Specimens Examined: Canada. Nunavut: Kitikmeot Region: Coppermine [Kugluktuk], near old Eskimo [sic] site above Anglican Mission [67°49′36″N, 115°5′36″W ± 1.5 km], 7 July 1958, R. D. Wood s.n. (CAN-265596); flats on W side of Fockler Creek, above spruce forest in creek valley, ca. 2.2 km S of Sandstone Rapids, Coppermine River, 67°25′49″N, 115°37′55″W ± 50 m, 152 m, 1 July 2014, Saarela, Sokoloff & Bull 3120 (CAN, UBC); S of Fockler Creek, along small tributary that runs into Fockler Creek, ca. 2.3 km S of Sandstone Rapids, Coppermine River, 67°25′44.9″N, 115°38′25.9″W ± 100 m, 152 m, 3 July 2014, Saarela, Sokoloff & Bull 3243 (ALTA, CAN); flats atop and upper slopes of Coppermine Mountains, N/W side of Coppermine River, 67°14′43.7″N, 115°38′51.2″W ± 150 m, 422 m, 9 July 2014, Saarela, Sokoloff & Bull 3748 (CAN, O); Kugluk (Bloody Falls) Territorial Park, NE-facing slope of large hill just S of Bloody Falls, W side of Coppermine River, 67°44′6.6″N, 115°23′13.4″W ± 50 m, 40 m, 14 July 2014, Saarela, Sokoloff & Bull 3982 (CAN); Heart Lake, SW of Kugluktuk, 6.4 km SW of mouth of Coppermine River, 67°48′6.7″N, 115°13′40.6″W ± 50 m, 41 m, 23 July 2014, Saarela, Sokoloff & Bull 4304 (CAN, MO, MT, US); W of Kugluktuk on tundra flats above Coppermine River, S of 1 Coronation Drive and N of power plant, 67°49′28.97″N, 115°5′0.2″W ± 100 m, 8 m, 25 July 2014, Saarela, Sokoloff & Bull 4373 (CAN).

Luzula groenlandica Böcher—Greenland woodrush | North American (N)

We found this species growing west of Kugluktuk along a trail where it was locally common. It is similar to Luzula multiflora subsp. frigida (Buchenau) V. I. Krecz., which has a sympatric range across the mainland Canadian Arctic (Porsild & Cody, 1980; Brooks & Clemants, 2002). The collection Findlay 246 at DAO from Kugluktuk, published in Cody (1954b) as Luzula nivalis, was reported as Luzula groenlandica in Cody & Porsild (1968) and mapped as such in Porsild & Cody (1980). However, a duplicate of Findlay 246 (ALTA-VP-4743) is determined as Luzula multiflora subsp. frigida. We were unable to locate the DAO sheet, and have not seen the ALTA sheet. If Findlay’s collection is confirmed as Luzula multiflora subsp. frigida it would be the first record of the taxon for the study area, and our collection would be the first confirmed record of Luzula groenlandica for the study area. Luzula groenlandica is not recorded for the Canadian Arctic Archipelago and is at the northern edge of its range in the study area. It has a scattered distribution across mainland Nunavut and there are several records from northern Quebec and Labrador (Porsild & Cody, 1980; Hay & Payette, 2013).

Specimens Examined: Canada. Nunavut: Kitikmeot Region: along ATV trail between Coronation Gulf and road from Kugluktuk to Heart Lake cemetery, 67°49′37.8″N, 115°10′31.8″W ± 500 m, 12 m, 23 July 2014, Saarela, Sokoloff & Bull 4339 (CAN, UBC).

Luzula nivalis (Laest.) Spreng.—Arctic wood rush | Circumpolar-alpine

Previously recorded from the study area (Cody, 1954b; Porsild & Cody, 1980) and widespread throughout the Canadian Arctic (Porsild & Cody, 1980; Korol, 1992; Aiken et al., 2007; Hay & Payette, 2013; Saarela et al., 2013a). We made collections at Fockler Creek, Melville Creek and Kugluk (Bloody Falls) Territorial Park.

Specimens Examined: Canada. Nunavut: Kitikmeot Region: Coppermine [Kugluktuk], Coronation Gulf, at mouth of Coppermine River [67.822146°N, 115.078387°W ± 0.5 km], 4 August 1948, H. T. Shacklette 3280 (CAN-199929); Coppermine [Kugluktuk], 67°49′36″N, 115°5′36″W, 6 August 1951, W. I. Findlay 261 (DAO-176296 01-01000677830); S of Fockler Creek, along small tributary that runs into Fockler Creek, ca. 2.3 km S of Sandstone Rapids, Coppermine River, 67°25′44.9″N, 115°38′25.9″W ± 100 m, 152 m, 3 July 2014, Saarela, Sokoloff & Bull 3246 (ALA, CAN); confluence of Coppermine River and Melville Creek, just W of Coppermine Mountains, 67°15′52″N, 115°30′55.3″W ± 350 m, 178–190 m, 7 July 2014, Saarela, Sokoloff & Bull 3518 (ALTA, CAN); Kugluk (Bloody Falls) Territorial Park, flats above boardwalk W of Bloody Falls, 67°44′34.5″N, 115°22′27″W ± 100 m, 135 m, 16 July 2014, Saarela, Sokoloff & Bull 4059 (CAN, MO, MT, O).

Juncaginaceae [1/2]

Triglochin maritima L.—Sea-side arrow-grass | Circumboreal–polar | Noteworthy Record

Newly recorded for the study area, where the species is at the northern edge of its range. We made collections at Fockler Creek, Kugluk (Bloody Falls) Territorial Park and Kugluktuk. Porsild & Cody (1980) recorded its distribution as barely extending north of the treeline and also mapped a site from the west side of Bathurst Inlet (Kelsall & McEwen 107, CAN-202950; Kelsall & McEwan 47, CAN-202949), an area where Bennett (2015) recently recorded it. Gould & Walker (1997) recorded it along the lower Hood River. Haynes & Hellquist (2000a) included southeastern Nunavut in its range, where there are a few records (Porsild, 1950b; Cody, 1996a; Cody & Reading, 2005). It was previously recorded from ca. 30 km west of the Coppermine River, just outside the study area (Reading 153, DAO; Cody & Reading, 2005). Further west, it is known from several sites in Tuktut Nogait National Park and vicinity (Cody, Scotter & Zoltai, 1992; Saarela et al., 2013a) and the Great Bear Lake area (Porsild, 1943; Porsild & Cody, 1980). It is not recorded for the Canadian Arctic Archipelago and is recorded for a few southern Arctic sites in northern Quebec and Labrador (Hay, 2013).

Specimens Examined: Canada. Nunavut: Kitikmeot Region: sedge meadow at S end of small lake, on flats NW of Fockler Creek, ca. 1.9 km SSE of Sandstone Rapids, Coppermine River, 67°26′1.8″N, 115°37′30.5″W ± 20 m, 170 m, 2 July 2014, Saarela, Sokoloff & Bull 3231 (CAN, UBC); Kugluk (Bloody Falls) Territorial Park, rocky valley immediately SW of Bloody Falls, along rough marked section of Portage Trail, head of small unnamed pond just W of falls, 67°44′42.8″N, 115°22′29.2″W ± 10 m, 9 m, 13 July 2014, Saarela, Sokoloff & Bull 3906 (CAN); W of Kugluktuk on tundra flats above Coppermine River, S of 1 Coronation Drive and N of community power plant, 67°49′28.97″N, 115°5′0.2″W ± 100 m, 8 m, 22 July 2014, Saarela, Sokoloff & Bull 4271 (ALA, CAN).

Triglochin palustris L.—Marsh arrow-grass | Circumboreal–polar | Noteworthy Record

Newly recorded for the study area and western Nunavut. We made collections in wet habitats (seeps, pond edges, upper edges of estuary) at Fockler Creek, Richardson Bay and Kugluk (Bloody Falls) Territorial Park. The nearest collections to the west are from the vicinity of Tuktut Nogait National Park (Saarela et al., 2013a). This species has a scattered distribution in Nunavut. In earlier works it was recorded for mainland Nunavut based on a record from Mistake Bay on the northwestern shore of Hudson Bay (Porsild, 1943; Porsild & Cody, 1980). This collection is not mapped in Haynes & Hellquist (2000a) even though the specimen was confirmed as this species by C. Hellquist prior to that publication. It has since been recorded from Bathurst Inlet (Scotter and Zoltai 31526, DAO; Cody, Scotter & Zoltai, 1984; Bennett, 2015) and the lower Hood River (Gould & Walker, 1997). There are other unpublished collections at CAN from Bathurst Inlet (Kelsall & McEwen 249, CAN-202952; Kelsall & McEwen 121, CAN-202951) not mapped in Porsild & Cody (1980). Elsewhere in Nunavut it was recently reported from Baffin Island, the first records for the Canadian Arctic Archipelago (Gillespie et al., 2015). Also known from Akimiski Island (Blaney & Kotanen, 2001) and northern Quebec and Labrador (Hay, 2013).

Specimens Examined: Canada. Nunavut: Kitikmeot Region: S of Fockler Creek, along small tributary that runs into Fockler Creek, ca. 2.3 km S of Sandstone Rapids, Coppermine River, 67°25′44.9″N, 115°38′25.9″W ± 100 m, 152 m, 3 July 2014, Saarela, Sokoloff & Bull 3258 (CAN); Richardson Bay, confluence of Richardson and Rae rivers at Coronation Gulf, ca. 20 km WNW of Kugluktuk, 67°54′11.2″N, 115°32′27.4″W ± 200 m, 0 m, 8 July 2014, Saarela, Sokoloff & Bull 3672 (CAN); Kugluk (Bloody Falls) Territorial Park, rocky valley immediately SW of Bloody Falls, along rough marked section of Portage Trail, 67°44′34″N, 115°22′16″W ± 50 m, 20 m, 13 July 2014, Saarela, Sokoloff & Bull 3861 (CAN); Kugluk (Bloody Falls) Territorial Park, rocky beach above Bloody Falls, W bank of Coppermine River, 67°44′18″N, 115°22′57.3″W ± 250 m, 34 m, 14 July 2014, Saarela, Sokoloff & Bull 3975 (ALTA, CAN); Kugluk (Bloody Falls) Territorial Park, rocky sandy beach just below Bloody Falls, W side of Coppermine River, vicinity of confluence with small creek, beach seasonally flooded, 67°44′54.5″N, 115°22′17.2″W ± 75 m, 9 m, 17 July 2014, Saarela, Sokoloff & Bull 4129 (CAN).

Orchidaceae [2/2]

Corallorhiza trifida Châtel., Fig. S10—Northern coralroot, early coralroot | Circumboreal–polar

Previously recorded from Kugluktuk (Cody, 1954b; Porsild & Cody, 1980). We made collections at Melville Creek, Richardson Bay, Fockler Creek, Kugluk (Bloody Falls) Territorial Park and Kugluktuk. The species was uncommon (only a few plants) where it was found. Elsewhere on mainland Nunavut recorded from numerous sites (Porsild & Cody, 1980; Cody, Reading & Line, 2003). In the Canadian Arctic Archipelago known from southern Baffin Island and recently recorded from Victoria Island north of the study area (Aiken et al., 2007; Gillespie et al., 2015). It is also recorded for Arctic areas of mainland Northwest Territories and northern Quebec and Labrador (Porsild & Cody, 1980; Houle, 2013; Saarela et al., 2013a).

Specimens Examined: Canada. Nunavut: Kitikmeot Region: Coppermine [Kugluktuk], 67°49′36″N, 115°5′36″W, 22 July 1951, W. I. Findlay 165 (DAO-176782 01-01000520765); Coppermine [Kugluktuk], Cemetery Island, 67°50′N, 115°7′W, 14 July 1951, W. I. Findlay 119 (DAO-176765 01-01000520764); slopes on E side of Coppermine River, N of its confluence with Fockler Creek, ca. 0.8 km SW of Sandstone Rapids, 67°26′36.9″N, 115°38′50.1″W ± 50 m, 128 m, 4 July 2014, Saarela, Sokoloff & Bull 3383 (CAN); confluence of Coppermine River and Melville Creek, just W of Coppermine Mountains, 67°15′52″N, 115°30′55.3″W ± 350 m, 178–190 m, 7 July 2014, Saarela, Sokoloff & Bull 3536 (CAN); Richardson Bay, confluence of Richardson and Rae rivers at Coronation Gulf, ca. 20 km WNW of Kugluktuk, 67°54′11.2″N, 115°32′27.4″W ± 200 m, 0 m, 8 July 2014, Saarela, Sokoloff & Bull 3684 (CAN); Kugluk (Bloody Falls) Territorial Park, along Portage Trail at top of ridge on W bank of Coppermine River, near start of Bloody Falls rapids, 67°44′22.5″N, 115°22′40.6″W ± 10 m, 46 m, 14 July 2014, Saarela, Sokoloff & Bull 3931 (CAN); Kugluk (Bloody Falls) Territorial Park, rocky valley immediately SW of Bloody Falls, along rough marked section of Portage Trail, 67°44′34″N, 115°22′16″W ± 50 m, 20 m, 18 July 2014, Saarela, Sokoloff & Bull 4159 (CAN); NW-facing moist to wet sedge meadow, drainage running into Coppermine River above Bloody Falls on SE side, 67°44′26.2″N, 115°22′11.8″W ± 15 m, 47 m, 19 July 2014, Saarela, Sokoloff & Bull 4193 (CAN); W of Kugluktuk on tundra flats above Coppermine River, S of 1 Coronation Drive and N of power plant, 67°49′28.97″N, 115°5′0.2″W ± 100 m, 8 m, 25 July 2014, Saarela, Sokoloff & Bull 4389 (CAN).

Platanthera obtusata (Banks ex Pursh) Lindl. subsp. obtusata, Fig. S11—Northern rein orchid, blunt leaved rein orchid | North American

Previously recorded from Kugluktuk (Cody, 1954b; Porsild & Cody, 1980). We made collections at Fockler Creek, Melville Creek, south of Escape Rapids, Kugluk (Bloody Falls) Territorial Park and Kugluktuk. Populations generally comprised scattered individuals growing in low shrub tundra and wet sedge meadows. Porsild & Cody (1980) noted this species is pollinated by mosquitoes, and that mosquitoes are often seen with pollinia attached to their heads. Indeed, we observed mosquitoes with pollinia from this species on numerous occasions in Kugluk (Bloody Falls) Territorial Park. This boreal species reaches the Low Arctic on the mainland (Porsild & Cody, 1980; Gould & Walker, 1997; Cody, Reading & Line, 2003; Houle, 2013; Saarela et al., 2013a; Bennett, 2015) and southern Baffin Island, where it was recently recorded (Gillespie et al., 2015). It is not known from the rest of the Canadian Arctic Archipelago (Aiken et al., 2007) and is at the northern edge of its range in the study area. It was recognised as Habenaria obtusata Banks ex Pursh in Porsild & Cody (1980) and Lysiella obtusata (Banks ex Pursh) Rydb. in Elven et al. (2011). Molecular work supports its placement in Platanthera Rich. (Bateman et al., 2009). All North American plants are recognised as Platanthera obtusata subsp. obtusata, while Eurasian plants are recognised as Platanthera obtusata subsp. oligantha (Turcz.) Hultén (Sheviak, 2003) or as a species, Lysiella oligantha (Turcz.) Nevski (=Platanthera oligantha Turcz.) (Elven et al., 2011).

Another Platanthera species, the boreal Platanthera aquilonis Sheviak—treated as H. hyperborea (L.) R. Br. in Porsild & Cody (1980), a taxon now restricted to Greenland (Sheviak, 2003)—was recently recorded from the nearby Big Bend area of the Coppermine River (Reading 32-1, DAO; Cody, Reading & Line, 2003). Although not stated in that publication, that report was the first for mainland Nunavut. It is also recorded for Akimiski Island in Nunavut (Blaney & Kotanen, 2001).

Specimens Examined: Canada. Nunavut: Kitikmeot Region: Coppermine [Kugluktuk], 67°49′36″N, 115°5′36″W, 4 August 1951, W. I. Findlay 251 (DAO-176720 01-01000677359); Coppermine [Kugluktuk], 67°49′36″N, 115°5′36″W, 24 July 1951, W. I. Findlay 181 (DAO-176719 01-01000677809); Coppermine [Kugluktuk], 67°49′36″N, 115°5′36″W, 2 August 1995, T. Dolman 95 (LEA); S side of Fockler Creek, ca. 2.7 SE of Sandstone Rapids, Coppermine River, 67°25′38.2″N, 115°36′54.9″W ± 50 m, 128 m, 5 July 2014, Saarela, Sokoloff & Bull 3400 (CAN); tundra below Tundra Lake and Fockler Creek, ca. 4.1 km SE of Sandstone Rapids, Coppermine River, 67°25′20.7″N, 115°34′17.2″W ± 25 m, 271 m, 5 July 2014, Saarela, Sokoloff & Bull 3427 (CAN, UBC); confluence of Coppermine River and Melville Creek, just W of Coppermine Mountains, 67°15′52″N, 115°30′55.3″W ± 350 m, 178–190 m, 7 July 2014, Saarela, Sokoloff & Bull 3535 (ALA, CAN); S-facing slopes on W side of Coppermine River, about halfway between Escape Rapids and Muskox Rapids, 67°31′18.2″N, 115°36′20.1″W ± 150 m, 115 m, 8 July 2014, Saarela, Sokoloff & Bull 3620 (ALTA, CAN); Kugluk (Bloody Falls) Territorial Park, rocky valley immediately SW of Bloody Falls, along rough marked section of Portage Trail, 67°44′34″N, 115°22′16″W ± 50 m, 20 m, 13 July 2014, Saarela, Sokoloff & Bull 3866 (CAN, O); W of Kugluktuk on tundra flats above Coppermine River, S of 1 Coronation Drive and N of community power plant, 67°49′28.97″N, 115°5′0.2″W ± 100 m, 8 m, 22 July 2014, Saarela, Sokoloff & Bull 4247 (CAN, MO).

Poaceae [17/39]

Agrostis mertensii Trin.—Northern bentgrass | Amphi-Pacific–North American (N)–amphi-Atlantic–European (N) | Noteworthy Record

Our single collection of this species from the Bigtree River, near the southern limit of the study area, represents a northeastern range extension from the nearest collections from Great Slave Lake (Porsild & Cody, 1980; Harvey, 2007). It was common in sandy soil along the river, growing with Alnus alnobetula, Betula glandulosa, Bromus pumpellianus, Dasiphora fruticosa, Picea glauca and Salix alaxensis. Elsewhere in the Canadian Arctic recorded from the Hood River, as Agrostis borealis Hartm. (Gould & Walker, 1997), several sites on the southeastern mainland of Nunavut, southern Baffin Island and northern Quebec and Labrador (Porsild, 1950b; Porsild & Cody, 1980).

Specimens Examined: Canada. Nunavut: Kitikmeot Region: confluence of Coppermine and Bigtree rivers, 66°56′23.8″N, 116°21′3.2″W ± 100 m, 265 m, 7 July 2014, Saarela, Sokoloff & Bull 3601 (ALA, ALTA, CAN, MT, O, UBC, US).

Alopecurus borealis Trin., Fig. S12—Alpine foxtail | Circumpolar-alpine

Previously recorded from Kugluktuk and Escape Rapids (Cody, 1954b; Porsild & Cody, 1980). We made collections at Richardson Bay, Kugluk (Bloody Falls) Territorial Park and Kugluktuk. Widespread throughout the Canadian Arctic (Porsild & Cody, 1980; Korol, 1992; Aiken et al., 2007). Taxonomy follows Elven et al. (2011), who recognised plants sometimes treated as the single polymorphic species Alopecurus magellanicus Lam. (type from Chile) (Soreng, 2003a; Crins, 2007), as multiple species: Alopecurus borealis, Alopecurus glaucus Less., Alopecurus rozhevitzianus Ovcz. and Alopecurus stejnegeri Vasey. This taxon was treated as Alopecurus alpinus Sm. (published in 1803) in Porsild & Cody (1980); that name is illegitimate (non Alopecurus alpinus Vill. 1786) and the next available one is Alopecurus borealis (Elven et al., 2011), described from “Asia et America boreali”. See Dogan (1999) for further taxonomic information.

Specimens Examined: Canada. Nunavut: Kitikmeot Region: Coppermine [Kugluktuk] [67°49′36″N, 115°5′36″W ± 1.5 km], 25 August 1934, A. H. Dutilly 160 (CAN-513320); Coppermine [Kugluktuk], 67°49′36″N, 115°5′36″W, 8 July 1951, W. I. Findlay 96 (ALTA-VP-1323, DAO-174384 01-01000677957, QFA-0232183, UBC-V40766); Rae River, s.d., J. Richardson s.n. (CAN-30525); Coppermine [Kugluktuk] [67°49′36″N, 115°5′36″W ± 1.5 km], 8 July 1958, R. D. Wood s.n. (CAN-265604); Escape Rapids, s.d., s.c. [Richardson?] (CAN-229874); Richardson Bay, confluence of Richardson and Rae rivers at Coronation Gulf, ca. 20 km WNW of Kugluktuk, 67°54′11.2″N, 115°32′27.4″W ± 200 m, 0 m, 8 July 2014, Saarela, Sokoloff & Bull 3678 (CAN, MO, NY, QFA, US, WIN); Kugluk (Bloody Falls) Territorial Park, rocky cliffs and ledges directly above (W side) of Bloody Falls, just S of heavily used day-use/fishing area, 67°44′40.1″N, 115°22′4.9″W ± 20 m, 8 m, 12 July 2014, Saarela, Sokoloff & Bull 3806 (ALA, CAN, US); clay slopes and beach on E side of Coppermine River, just above start of Bloody Falls, 67°44′9.4″N, 115°22′41.2″W ± 15 m, 40 m, 19 July 2014, Saarela, Sokoloff & Bull 4216 (ALTA, CAN, O); grassy vacant lot in Kugluktuk, 67°49′30.5″N, 115°5′29.3″W ± 15 m, 21 m, 24 July 2014, Saarela, Sokoloff & Bull 4346 (CAN, MT, UBC).

Anthoxanthum arcticum Veldkamp—Few-flowered arctic holy grass | Asian (N)–amphi-Beringian–North American (N) | Noteworthy Record

Newly recorded for the study area. Our single collection represents a southern range extension in the central part of the species’ North American range, which primarily comprises the Canadian Arctic Archipelago (Aiken et al., 2007). On the mainland to the east known from the Hood River (Gould & Walker, 1997) and shores of Hudson Bay in Nunavut, Manitoba and Ontario, and to the west from Tuktut Nogait National Park and vicinity (Saarela et al., 2013a) and the Mackenzie Delta area (Porsild & Cody, 1980; Korol, 1992; Allred & Barkworth, 2007). Our collection was gathered in a mesic sedge meadow just outside the southwestern boundary of Kugluk (Bloody Falls) Territorial Park, on a large upland plateau. It was locally common and growing with Arctagrostis latifolia subsp. latifolia, Carex membranacea, Eriophorum triste and Eriophorum vaginatum. This species has traditionally been recognised as Hierochloe pauciflora R. Br. Many recent authors (Soreng, 2003b; Allred & Barkworth, 2007)—but not all of them (Elven et al., 2011)—include Hierochloe R. Br in Anthoxanthum L., a classification supported by molecular data (Pimentel et al., 2013).

Specimens Examined: Canada. Nunavut: Kitikmeot Region: flats on top of mountain on W side of Coppermine River, just S of the start of Bloody Falls Rapids, 67°43′53.7″N, 115°24′31.4″W ± 3 m, 112 m, 14 July 2014, Saarela, Sokoloff & Bull 4001 (CAN, UBC, US).

Anthoxanthum hirtum (Schrank) Y. Schouten & Veldkamp—Hairy sweetgrass | Circumboreal-polar | Noteworthy Record

Newly recorded for the study area and our single collection represents a range extension. It was locally common on a south-facing slope above the Coppermine River just outside Kugluktuk, growing with Arnica angustifolia, Chamerion angustifolium, Poa glauca subsp. glauca and Potentilla sp. The nearest record to the east is from the Hope Bay area northeast of Bathurst Inlet (Reading 317, DAO; Cody & Reading, 2005), a site well beyond the main range of the species and the only other record on mainland Nunavut (Allred & Barkworth, 2007). Elsewhere in Nunavut recorded from Akimiski Island (Blaney & Kotanen, 2001) and other islands in Hudson Bay (Allred & Barkworth, 2007). It is not recorded for the Canadian Arctic Archipelago. The nearest collections to the southwest are from Great Slave Lake (Porsild & Cody, 1980) and to the west from the lower Brock River area. The latter collection was made by our team in 2009, but was accidentally omitted from Saarela et al. (2013a); it is reported here. This taxon was treated as Hierochloë odorata (L.) Beauv. in Porsild & Cody (1980). In Elven et al. (2011) the plants fall under Hierochloë odorata subsp. arctica (J. Presl) Tzvelev (syn. Anthoxanthum hirtum subsp. arcticum (J. Presl) G. C. Tucker).

Specimens Examined: Canada. Northwest Territories: vicinity of the Lower Brock River, west of Tuktut Nogait National Park, coastal area ca. 10 km N of Brock River delta, 69°36′16″N, 123°09′04″W, 0–10 m, Gillespie, Consaul and Bull 9404 (CAN-594920). Nunavut: Kitikmeot Region: W of Kugluktuk on tundra flats above Coppermine River, S of 1 Coronation Drive and N of community power plant, 67°49′28.97″N, 115°5′0.2″W ± 100 m, 8 m, 22 July 2014, Saarela, Sokoloff & Bull 4260 (ALA, ALTA, CAN, UBC, US).

Anthoxanthum monticola subsp. alpinum (Sw. ex Willd.) Soreng—Alpine sweet grass | Circumpolar-alpine

Previously recorded from Kugluktuk, as Hierochloe alpina (Sw.) Roem. & Schult. (Cody, 1954b; Porsild & Cody, 1980). We made collections of this acid-loving species at Fockler Creek, Kendall River, Kugluk (Bloody Falls) Territorial Park, Heart Lake and Kugluktuk. Widespread across the Canadian Arctic (Porsild & Cody, 1980; Cody, Scotter & Zoltai, 1989; Korol, 1992; Aiken et al., 2007; Saarela et al., 2013a; Bennett, 2015). Taxonomy follows Allred & Barkworth (2007) and Elven et al. (2011), who recognise two subspecies. The other one, Anthoxanthum monticola (Bigelow) Veldkamp subsp. monticola has a more easterly distribution, occurring as far west as the eastern side of Hudson Bay (Allred & Barkworth, 2007). Bennett (2015) recently reported subsp. monticola from sites on southern Victoria Island, representing a major range extension and the first records for the Canadian Arctic Archipelago.

Specimens Examined: Canada. Nunavut: Kitikmeot Region: Coppermine [Kugluktuk], 67°49′36″N, 115°5′36″W, 2 July 1951, W. I. Findlay 70 (DAO-174258 01-01000677489); Coppermine [Kugluktuk], 67°49′36″N, 115°5′36″W, 4 July 1951, W. I. Findlay 89 (DAO-174257 01-01000677485); Kugluktuk, airport, 67.816667°N, 115.143889°W, 2 July 2008, L. J. Gillespie, J. M. Saarela, L. M. Consaul & R. D. Bull 7456 (CAN-592317); Coppermine [Kugluktuk], ridge above Eskimo [sic] camp west of DOT [67°49′36″N, 115°5′36″W ± 1.5 km], 2 July 1958, R. D. Wood s.n. (CAN-265603); Kugluktuk, rocky slopes of North Hill, 67°49′31.4″N, 115°6′54″W ± 100 m, 42 m, 29 June 2014, Saarela, Sokoloff & Bull 3079 (CAN, O); flats on W side of Fockler Creek, above spruce forest in creek valley, ca. 2.2 km S of Sandstone Rapids, Coppermine River, 67°25′49″N, 115°37′55″W ± 50 m, 152 m, 1 July 2014, Saarela, Sokoloff & Bull 3112 (CAN, NY, QFA, US, WIN); confluence of Coppermine and Kendall rivers (NW side of Coppermine River, S side of Kendall River), 67°6′51.1″N, 116°8′18.3″W ± 150 m, 220 m, 7 July 2014, Saarela, Sokoloff & Bull 3571 (CAN, MO); Kugluk (Bloody Falls) Territorial Park, day-use area above Bloody Falls (at outhouse and fire pit), 67°44′36.8″N, 115°22′11.1″W ± 25 m, 28 m, 12 July 2014, Saarela, Sokoloff & Bull 3839 (ALA, ALTA, CAN, O, UBC, US); Heart Lake, SW of Kugluktuk, 6.4 km SW of mouth of Coppermine River, 67°48′6.7″N, 115°13′40.6″W ± 50 m, 41 m, 23 July 2014, Saarela, Sokoloff & Bull 4306 (ALA, ALTA, CAN, K, MO, MT, NY, O, QFA, UBC, US, WIN); W of Kugluktuk on tundra flats above Coppermine River, S of 1 Coronation Drive and N of power plant, 67°49′28.97″N, 115°5′0.2″W ± 100 m, 8 m, 25 July 2014, Saarela, Sokoloff & Bull 4374 (CAN, MT, WIN).

Arctagrostis latifolia subsp. arundinacea (Trin.) Tzvelev—Reed polargrass | Asian (N)–amphi-Beringian | Noteworthy Record

Newly recorded for the study area. Our collections close a distribution gap between the Brock Lagoon area, eastern Great Slave Lake and a site near the coast just west of Bathurst Inlet, the subspecies’ known eastern limit (Aiken & Lefkovitch, 1990; Saarela et al., 2013a). A record from southwestern Victoria Island is mapped in Aiken & Lefkovitch (1990), but not in Aiken et al. (2007), who did not record this subspecies for the Canadian Arctic Archipelago. We collected this subspecies only from Kugluktuk; we did not collect subsp. latifolia there. Taxonomy follows Aiken et al. (2007), whereas Elven et al. (2011) recognise Arctagrostis latifolia and Arctagrostis arundinacea (Trin.) Beal at species level.

Specimens Examined: Canada. Nunavut: Kitikmeot Region: W of Kugluktuk on tundra flats above Coppermine River, S of 1 Coronation Drive and N of community power plant, 67°49′28.97″N, 115°5′0.2″W ± 100 m, 8 m, 22 July 2014, Saarela, Sokoloff & Bull 4252 (CAN, US); Kugluktuk, roadside and flats between buildings, 67°49′27.4″N, 115°5′26.2″W ± 25 m, 29 m, 26 July 2014, Saarela, Sokoloff & Bull 4393 (ALA, ALTA, CAN, O, UBC).

Arctagrostis latifolia (R. Br.) Griseb. subsp. latifolia—Polargrass, Arctic grass | Circumpolar-alpine

Previously recorded from the Rae River (Porsild & Cody, 1980; Aiken & Lefkovitch, 1990). We made collections at Fockler Creek, Kugluk (Bloody Falls) Territorial Park, Heart Lake and west of Kugluktuk. This subspecies is widespread throughout the Canadian Arctic (Porsild & Cody, 1980; Cody, Scotter & Zoltai, 1989; Aiken & Lefkovitch, 1990; Korol, 1992; Aiken et al., 2007; Saarela et al., 2013a). See comments regarding taxonomy under subsp. arundinacea.

Specimens Examined: Canada. Nunavut: Kitikmeot Region: Rae River [67°55′N, 115°30′W], Richardson s.n. (CAN-30955, mixed with Calamagrostis stricta subsp. groenlandica); old riverbed of Fockler Creek, ca. 2.3 km SSE of Sandstone Rapids, Coppermine River, 67°25′45.7″N, 115°37′21.8″W ± 25 m, 166 m, 1 July 2014, Saarela, Sokoloff & Bull 3168 (CAN, MO); ridge N of Fockler Creek, ca. 2.1 km SSE of Sandstone Rapids, Coppermine River, 67°25′54″N, 115°37′30″W ± 25 m, 166 m, 2 July 2014, Saarela, Sokoloff & Bull 3207 (CAN, K, NY); N side of Fockler Creek, ca. 1.9 km S of Sandstone Rapids, Coppermine River, 67°25′57.89″N, 115°38′3.9″W ± 10 m, 162 m, 4 July 2014, Saarela, Sokoloff & Bull 3323 (CAN, MT); Kugluk (Bloody Falls) Territorial Park, along Portage Trail at top of ridge on W bank of Coppermine River, near start of Bloody Falls rapids, 67°44′22.5″N, 115°22′40.6″W ± 10 m, 46 m, 14 July 2014, Saarela, Sokoloff & Bull 3929 (CAN, WIN); Heart Lake, SW of Kugluktuk, 6.4 km SW of mouth of Coppermine River, 67°48′6.7″N, 115°13′40.6″W ± 50 m, 41 m, 23 July 2014, Saarela, Sokoloff & Bull 4302 (ALA, CAN, UBC, US); manufactured gravel slopes around Kugluktuk’s sewage retention pond, 5.16 km SW of Coppermine River, 67°48′52.38″N, 115°12′10.3″W ± 10 m, 35 m, 23 July 2014, Saarela, Sokoloff & Bull 4331 (CAN, QFA).

Arctophila fulva (Trin.) Andersson, Fig. S13—Pendent grass | Circumpolar

Previously recorded from Bloody Falls, as Colpodium fulvum (Trin.) Griseb. (Cody, 1954b), but not mapped for the area in Porsild & Cody (1980) or Cayouette & Darbyshire (2007). We made collections of this aquatic grass at Kendall River, Fockler Creek, Expeditor Cove, Kugluk (Bloody Falls) Territorial Park and Kugluktuk. Elsewhere in the Canadian Arctic recorded from several Arctic islands (but not known in the High Arctic) and scattered mainland sites (Porsild & Cody, 1980; Korol, 1992; Aiken et al., 2007; Saarela et al., 2013a).

Specimens Examined: Canada. Nunavut: Kitikmeot Region: Bloody Falls on Coppermine River, 67°44′N, 115°23′W, 27 July 1951, W. I. Findlay 207 (ACAD-30930, DAO-172598 01-01000684828, QFA0232871, UBC-V40764); confluence of Coppermine and Kendall rivers (NW side of Coppermine River, S side of Kendall River), small ponds on flats adjacent to Coppermine River, 67°6′44.7″N, 116°8′6.1″W ± 100 m, 213 m, 7 July 2014, Saarela, Sokoloff & Bull 3587 (ALA, CAN, UBC); E end of small, unnamed lake on W bank of Coppermine River, ca. 8.3 km NNE of Sandstone Rapids, 67°31′30.8″N, 115°36′16.1″W ± 50 m, 126 m, 8 July 2014, Saarela, Sokoloff & Bull 3659 (CAN, K, MT, NY, QFA, UBC, WIN); Coronation Gulf, NW peninsula of Expeditor Cove, ca. 9.6 km NW of Kugluktuk, 67°52′47.2″N, 115°16′40.3″W ± 3 m, 17 m, 8 July 2014, Saarela, Sokoloff & Bull 3699 (CAN, US); Kugluk (Bloody Falls) Territorial Park, confluence of small unnamed creek and W bay of Coppermine River, rocky beach just below Bloody Falls, 67°44′45.3″N, 115°22′19.6″W ± 4 m, 12 m, 13 July 2014, Saarela, Sokoloff & Bull 3909 (ALTA, CAN, MT, O, WIN); Kugluk (Bloody Falls) Territorial Park, rocky sandy beach just below Bloody Falls, W side of Coppermine River, vicinity of confluence with small creek, beach seasonally flooded, 67°44′54.5″N, 115°22′17.2″W ± 75 m, 9 m, 17 July 2014, Saarela, Sokoloff & Bull 4112 (ALA, ALTA, CAN, MO, MT, O, UBC, US); N side of Heart Lake, below rocky cliff, SW of Kugluktuk, 5.64 km SW of mouth of Coppermine River, 67°48′33.8″N, 115°12′52.9″W ± 15 m, 39 m, 23 July 2014, Saarela, Sokoloff & Bull 4314 (CAN, K, NY, QFA, UBC); W of Kugluktuk on tundra flats above Coppermine River, S of 1 Coronation Drive and N of power plant, 67°49′28.97″N, 115°5′0.2″W ± 100 m, 8 m, 25 July 2014, Saarela, Sokoloff & Bull 4369 (ALA, ALTA, CAN, MT, US).

Bromus pumpellianus Scribn.—Pumpelly’s brome grass | European (NE)–Asian (N/C)–amphi-Beringian–North American

Previously recorded from Kugluktuk and Bloody Falls (Cody, 1954b; Porsild & Cody, 1980). We made numerous collections at Fockler Creek, Melville Creek, Coppermine Mountains, Kugluk (Bloody Falls) Territorial Park and Kugluktuk. Not recorded from the Canadian Arctic Archipelago, but known on the mainland Arctic from several sites, occurring as far east as Bathurst Inlet and along the Kazan River (Porsild & Cody, 1980; Bennett, 2015). This species is at the northern edge of its range in the study area. Following Saarela (2008) we do not recognise infraspecific taxa in North America. Elven et al. (2011) treated it in the segregate genus Bromopsis (Dumort.) Fourr., a classification not supported by molecular data (Saarela et al., 2007). They recognised seven subspecies, of which only Bromus pumpellianus subsp. arctica (Shear) Á. Löve & D. Löve was recorded for the Canadian Arctic.

Specimens Examined: Canada. Nunavut: Kitikmeot Region: Coppermine [Kugluktuk], 25 August 1934, A. H. Dutilly 232 (CAN-513847); Bloody Falls, 67°44′N, 115°23′W, 27 July 1951, W. I. Findlay 208 (ACAD-30933, DAO-171833 01-01000677941, UBC-V40765); Kugluktuk Airport, near weather station, 67°49′1.3″N, 115°8′2.3″W ± 25 m, 27 m, 30 June 2014, Saarela, Sokoloff & Bull 3106 (ALA, CAN, K, MO, O); old riverbed of Fockler Creek, ca. 2.3 km SSE of Sandstone Rapids, Coppermine River, 67°25′48″N, 115°37′33″W ± 25 m, 153 m, 1 July 2014, Saarela, Sokoloff & Bull 3143 (CAN, US); spruce forest along Fockler Creek, ca. 2.3 km SSE of Sandstone Rapids, Coppermine River, 67°25′45.7″N, 115°37′21.8″W ± 25 m, 166 m, 2 July 2014, Saarela, Sokoloff & Bull 3205 (CAN, NY, QFA, WIN); N side of Fockler Creek, ca. 1.9 km S of Sandstone Rapids, Coppermine River, 67°25′57.89″N, 115°38′3.9″W ± 10 m, 162 m, 4 July 2014, Saarela, Sokoloff & Bull 3331 (CAN, UBC); confluence of Sleigh Creek and Coppermine River, 0.4 km N of Sandstone Rapids, 67°27′13.9″N, 115°38′7.7″W ± 25 m, 126 m, 6 July 2014, Saarela, Sokoloff & Bull 3483 (CAN, MT); confluence of Coppermine River and Melville Creek, just W of Coppermine Mountains, 67°15′52″N, 115°30′55.3″W ± 350 m, 178–190 m, 7 July 2014, Saarela, Sokoloff & Bull 3491 (ALTA, CAN, UBC, US); Kugluk (Bloody Falls) Territorial Park, rocky cliffs and ledges directly above (W side) of Bloody Falls, just S of heavily used day-use/fishing area, 67°44′40.1″N, 115°22′4.9″W ± 20 m, 8 m, 12 July 2014, Saarela, Sokoloff & Bull 3799 (CAN, MO, O); Kugluk (Bloody Falls) Territorial Park, sandy rocky beach at W side of Bloody Falls, 67°44′27.8″N, 115°22′20.3″W ± 25 m, 5 m, 13 July 2014, Saarela, Sokoloff & Bull 3912 (ALA, ALTA, CAN, UBC, US); W of Kugluktuk on tundra flats above Coppermine River, S of 1 Coronation Drive and N of community power plant, 67°49′28.97″N, 115°5′0.2″W ± 100 m, 8 m, 22 July 2014, Saarela, Sokoloff & Bull 4261 (ALA, ALTA, CAN).

Calamagrostis canadensis subsp. langsdorffii (Link) Hultén—Langsdorff’s reedgrass |Nearly circumboreal-polar | Noteworthy Record

Newly recorded for the study area, filling a distribution gap between Tuktut Nogait National Park, eastern Great Bear Lake, Hood River and Bathurst Inlet (Porsild & Cody, 1980; Gould & Walker, 1997; Saarela et al., 2013a; Bennett, 2015). We made collections at Melville Creek, Kugluk (Bloody Falls) Territorial Park and Kugluktuk. In the park this taxon was common in the day-use area, where it grew in the understory of dense willow (Salix glauca) thickets and formed dense patches in areas cleared of willows for ATV trails and picnic tables. Elsewhere in the Canadian Arctic recorded from southeastern mainland Nunavut, Bathurst Inlet, southern Baffin Island and one site in Tuktut Nogait National Park (Porsild, 1950b; Porsild & Cody, 1980; Aiken et al., 2007; Saarela et al., 2013a; Bennett, 2015). It is at the northern edge of its range in the study area. This northern race of the widespread Calamagrostis canadensis (Michx.) P. Beauv. is recognised as var. langsdorffii (Link) Inman in Marr, Hebda & Greene (2007).

Specimens Examined: Canada. Nunavut: Kitikmeot Region: confluence of Coppermine River and Melville Creek, just W of Coppermine Mountains, 67°15′52″N, 115°30′55.3″W ± 350 m, 178–190 m, 7 July 2014, Saarela, Sokoloff & Bull 3514 (CAN, US); Kugluk (Bloody Falls) Territorial Park, day-use area above Bloody Falls (at outhouse and fire pit), 67°44′36.8″N, 115°22′11.1″W ± 25 m, 28 m, 20 July 2014, Saarela, Sokoloff & Bull 4232 (CAN, UBC); SE edge of Kugluktuk, rocky cliffs overlooking Coppermine River, 67°49′9.2″N, 115°5′40.4″W ± 50 m, 28 m, 24 July 2014, Saarela, Sokoloff & Bull 4360 (ALA, ALTA, CAN).

Calamagrostis lapponica (Wahlenb.) Hartm.—Lapland reedgrass | Circumboreal-polar | Noteworthy Record

Newly recorded from the study area, filling a distribution gap between eastern Great Bear Lake, Bathurst Inlet and September (Mouse) Lake, just west of the Coppermine River (Reading 115, DAO) (Porsild & Cody, 1980; Cody & Reading, 2005). We made collections at Fockler Creek, Kugluk (Bloody Falls) Territorial Park, Heart Lake and Kugluktuk. This grass is locally common in the disturbed day-use area of the park and is at the northern edge of its range in the study area. Elsewhere in Nunavut recorded from sites on the eastern mainland and near Iqaluit (Porsild, 1950b; Porsild & Cody, 1980; Cody & Reading, 2005; Aiken et al., 2007). Bennett (2015) reported an observation of this species from a site west of Hope Bay on southern Victoria Island, but did not make a collection. If confirmed, this species would be newly recorded for Victoria Island and the western Canadian Arctic Archipelago.

Specimens Examined: Canada. Nunavut: Kitikmeot Region: old riverbed of Fockler Creek, ca. 2.3 km SSE of Sandstone Rapids, Coppermine River, 67°25′48″N, 115°37′33″W ± 25 m, 153 m, 1 July 2014, Saarela, Sokoloff & Bull 3142 (CAN, US); Kugluk (Bloody Falls) Territorial Park, day-use area above Bloody Falls (at outhouse and fire pit), 67°44′36.8″N, 115°22′11.1″W ± 25 m, 28 m, 12 July 2014, Saarela, Sokoloff & Bull 3838 (CAN, UBC); Kugluk (Bloody Falls) Territorial Park, along wet, muddy, and deeply pitted ATV trail ca. 1 km W of Bloody Falls, 67°44′33.2″N, 115°23′30″W ± 20 m, 73 m, 16 July 2014, Saarela, Sokoloff & Bull 4101 (ALTA, CAN); Heart Lake, SW of Kugluktuk, 6.4 km SW of mouth of Coppermine River, 67°48′6.7″N, 115°13′40.6″W ± 50 m, 41 m, 23 July 2014, Saarela, Sokoloff & Bull 4301 (ALA, CAN, MT, O).

Calamagrostis purpurascens R. Br. subsp. purpurascens, Fig. S14—Purple reedgrass | Asian (NE)–amphi-Beringian–Cordilleran–North American–amphi-Atlantic (W)

Previously recorded from Kugluktuk (Cody, 1954b; Porsild & Cody, 1980). We made collections at Fockler Creek, Kugluk (Bloody Falls) Territorial Park and Kugluktuk. Elsewhere in the Canadian Arctic recorded from Baffin, Banks, Ellesmere, Melville and Victoria islands, a few sites on mainland Northwest Territories, and Bathurst Inlet (Porsild & Cody, 1980; Cody & Reading, 2005; Aiken et al., 2007; Saarela et al., 2013a; Bennett, 2015). Taxonomy follows Elven et al. (2011) who recognise two subspecies: the widespread subsp. purpurascens, and the northeastern North American subsp. laricina (Louis-Marie) Elven (also see Elven & Murray, 2008). Other authors do not recognise infraspecific taxa (Porsild & Cody, 1980; Marr, Hebda & Greene, 2007). This variation requires further study. This species was described by Robert Brown in Richardson (1823) based on material collected between Point Lake and the Arctic coast (i.e., along the Coppermine River), possibly from within the study area (type/possible type material BM001042267, NY00029601).

Specimens Examined: Canada. Nunavut: Kitikmeot Region: Coppermine [Kugluktuk], 67°49′36″N, 115°5′36″W, 31 July 1951, W. I. Findlay 217 (DAO-138225 01-01000677942); Kugluktuk, rocky slopes of North Hill, 67°49′31.4″N, 115°6′54″W ± 100 m, 42 m, 29 June 2014, Saarela, Sokoloff & Bull 3080 (CAN, US); flats on W side of Fockler Creek, above spruce forest in creek valley, ca. 2.2 km S of Sandstone Rapids, Coppermine River, 67°25′49″N, 115°37′55″W ± 50 m, 152 m, 1 July 2014, Saarela, Sokoloff & Bull 3126 (ALTA, CAN, O); second ridge N of Fockler Creek, ca. 1.9 km SSE of Sandstone Rapids, Coppermine River, 67°26′2.4″N, 115°37′26.5″W ± 25 m, 187 m, 2 July 2014, Saarela, Sokoloff & Bull 3214 (CAN, MO, MT); Coppermine River, sandstone cliffs above Sandstone Rapids, 67°27′29.6″N, 115°37′59.3″W ± 100 m, 110 m, 6 July 2014, Saarela, Sokoloff & Bull 3476 (CAN); Kugluk (Bloody Falls) Territorial Park, rocky cliffs and ledges directly above (W side) of Bloody Falls, just S of heavily used day-use/fishing area, 67°44′40.1″N, 115°22′4.9″W ± 20 m, 8 m, 12 July 2014, Saarela, Sokoloff & Bull 3800 (CAN); Kugluk (Bloody Falls) Territorial Park, S-facing cliff (gabbro sill) above start of Bloody Falls, W side of Coppermine River, W side of Portage Trail, 67°44′23.2″N, 115°22′54.5″W ± 50 m, 57 m, 16 July 2014, Saarela, Sokoloff & Bull 4078 (CAN, UBC); W of Kugluktuk on tundra flats above Coppermine River, S of 1 Coronation Drive and N of community power plant, 67°49′28.97″N, 115°5′0.2″W ± 100 m, 8 m, 22 July 2014, Saarela, Sokoloff & Bull 4262 (CAN, QFA, WIN); rocky cliffs on S side of Kugluktuk, 67°49′13″N, 115°5′55.8″W ± 50 m, 65 m, 26 July 2014, Saarela, Sokoloff & Bull 4402 (ALA, CAN).

Calamagrostis stricta subsp. groenlandica (Schrank) Á. Löve, Fig. 29—Slim-stemmed reedgrass | Circumpolar | Noteworthy Record

Figure 29 Calamagrostis stricta subsp. groenlandica.

(A) Inflorescence, Saarela et al. 4086. (B) Habit, Saarela et al. 4086. Photographs by P. C. Sokoloff.

Newly recorded for the study area. Our collections, gathered from Fockler Creek, Melville Creek and Kugluk (Bloody Falls) Territorial Park, close a distribution gap between southwestern Victoria Island, eastern Great Bear Lake and Bathurst Inlet (Porsild & Cody, 1980; Cody, Scotter & Zoltai, 1984; Aiken et al., 2007). The taxon was recognised as Calamagrostis neglecta G. Gaertn., B. Mey. & Scherb. s.l. by Porsild & Cody (1980) and Calamagrostis neglecta subsp. groenlandica (Schrank) Matuszk. by Elven et al. (2011) and Aiken et al. (2007), who recorded it in the Canadian Arctic Archipelago from Banks, Melville and Prince Patrick islands. A collection from Rae River (CAN-30955), gathered by J. Richardson, is part of a mixed sheet with Arctagrostis latifolia subsp. latifolia. The identification on the original label is Calamagrostis stricta. The Calamagrostis plant bears a note “too young for determination. Habit of Calamagrostis purpurascens”, in what appears to be Porsild’s hand. The specimen has short awns and better fits Calamagrostis stricta subsp. groenlandica. We cannot be sure that the specimen is from the mouth of the Rae River (and thus part of the study area), so it is not included here.

Specimens Examined: Canada. Nunavut: Kitikmeot Region: old riverbed of Fockler Creek, ca. 2.3 km SSE of Sandstone Rapids, Coppermine River, 67°25′45.7″N, 115°37′21.8″W ± 25 m, 166 m, 2 July 2014, Saarela, Sokoloff & Bull 3174 (CAN, US); confluence of Coppermine River and Melville Creek, just W of Coppermine Mountains, 67°15′52″N, 115°30′55.3″W ± 350 m, 178–190 m, 7 July 2014, Saarela, Sokoloff & Bull 3498 (CAN); Kugluk (Bloody Falls) Territorial Park, rocky cliffs and ledges directly above (W side) of Bloody Falls, just S of heavily used day-use/fishing area, 67°44′40.1″N, 115°22′4.9″W ± 20 m, 8 m, 12 July 2014, Saarela, Sokoloff & Bull 3797 (ALA, CAN); Kugluk (Bloody Falls) Territorial Park, rocky valley immediately SW of Bloody Falls, along rough marked section of Portage Trail, head of small unnamed pond just W of falls, 67°44′42.8″N, 115°22′29.2″W ± 10 m, 9 m, 13 July 2014, Saarela, Sokoloff & Bull 3905 (CAN); Kugluk (Bloody Falls) Territorial Park, S-facing cliff (gabbro sill) above start of Bloody Falls, W side of Coppermine River, W side of Portage Trail, 67°44′23.2″N, 115°22′54.5″W ± 50 m, 57 m, 16 July 2014, Saarela, Sokoloff & Bull 4086 (CAN).

Calamagrostis stricta (Timm) Koeler subsp. stricta—Slim-stemmed reedgrass | Circumboreal-polar | Noteworthy Record

Newly recorded for the study area. We made two collections from the river-side beach in Kugluk (Bloody Falls) Territorial Park, where it was growing with Artemisia tilesii, Chamerion latifolium, Equisetum arvense subsp. alpestre and Salix planifolia. These represent a minor range extension from a site ca. 30 km east of the Coppermine River in the Arctic ecozone (Reading 110, DAO; Cody & Reading, 2005). This subspecies is not recorded for the Canadian Arctic Archipelago, and is at the northern edge of its known range in the study area.

Specimens Examined: Canada. Nunavut: Kitikmeot Region: Kugluk (Bloody Falls) Territorial Park, rocky sandy beach just below Bloody Falls, W side of Coppermine River, vicinity of confluence with small creek, beach seasonally flooded, 67°44′54.5″N, 115°22′17.2″W ± 75 m, 9 m, 17 July 2014, Saarela, Sokoloff & Bull 4125 (CAN, UBC, US); Kugluk (Bloody Falls) Territorial Park, rocky sandy beach just below Bloody Falls, W side of Coppermine River, vicinity of confluence with small creek, beach seasonally flooded, 67°44′54.5″N, 115°22′17.2″W ± 75 m, 9 m, 17 July 2014, Saarela, Sokoloff & Bull 4132 (CAN).

Deschampsia brevifolia R. Br., Fig. 30A—Arctic hairgrass | Asian (N)–amphi-Beringian–North American (N)

Figure 30 Deschampsia brevifolia and Elymus alaskanus subsp. alaskanus.

Deschampsia brevifolia: (A) habit (bottom right), Saarela et al. 4098. Elymus alaskanus subsp. alaskanus: (B) habit, Saarela et al. 3191. (C) Habitat, Saarela et al. 3191. Photographs by P. C. Sokoloff.

Previously recorded from Bloody Falls (Cody, 1954b; Porsild & Cody, 1980). We encountered it at a single site in Kugluk (Bloody Falls) Territorial Park, along a wet and deeply pitted ATV trail (likely pitted due to melting permafrost) growing in luxuriant grass and sedge vegetation with Arctagrostis latifolia subsp. latifolia, Bistorta vivipara, Poa arctica subsp. arctica, Juncus leuchochlamys and Salix spp. Elsewhere on mainland Nunavut known from three other collections (Porsild & Cody, 1980; Cody, 1996a), and widespread throughout most of the Canadian Arctic Archipelago, but apparently rare on most of Baffin Island (Aiken et al., 2007). Chiapella et al. (2011) suggested Deschampsia brevifolia is best treated as an infraspecific taxon of Deschampsia cespitosa. As the name Deschampsia cespitosa subsp. brevifolia (Griseb.) Tzvelev is illegitimate (Elven et al., 2011), the combination Deschampsia cespitosa subsp. septentrionalis Chiapella was recently proposed (Chiapella, 2016).

Specimens Examined: Canada. Nunavut: Kitikmeot Region: Bloody Falls on Coppermine River, 67°44′N, 115°23′W, 27 July 1951, W. I. Findlay 206 (DAO-174161 01-01000679243); Kugluk (Bloody Falls) Territorial Park, along wet, muddy, and deeply pitted ATV trail ca. 1 km W of Bloody Falls, 67°44′33.2″N, 115°23′30″W ± 20 m, 73 m, 16 July 2014, Saarela, Sokoloff & Bull 4098 (CAN, UBC, US).

Deschampsia cespitosa (L.) P. Beauv. subsp. cespitosa—Tufted hairgrass | Circumboreal

Previously recorded from the study area (Porsild & Cody, 1980), but we were unable to locate a voucher specimen. We made collections at Fockler Creek, Coppermine Mountains, Kendall River and Kugluk (Bloody Falls) Territorial Park. One collection placed here (no. 4217) is intermediate between Deschampsia cespitosa subsp. cespitosa and Deschampsia brevifolia, following the circumscriptions in Barkworth (2007). The plant has narrow inflorescences (ca. 2 cm wide at anthesis) with spikelets densely clustered at the ends of the branches, like Deschampsia brevifolia, and glumes and lemmas purple proximally for less than one-half of their surface, usually with a green base, like subsp. cespitosa. The plants have dehiscing anthers and thus are not considered to be a hybrid.

Specimens Examined: Canada. Nunavut: Kitikmeot Region: confluence of Sleigh Creek and Coppermine River, 0.4 km N of Sandstone Rapids, 67°27′13.9″N, 115°38′7.7″W ± 25 m, 126 m, 6 July 2014, Saarela, Sokoloff & Bull 3480 (ALTA, CAN, MO, MT, O, UBC, US); confluence of Coppermine River and Melville Creek, just W of Coppermine Mountains, 67°15′52″N, 115°30′55.3″W ± 350 m, 178–190 m, 7 July 2014, Saarela, Sokoloff & Bull 3506 (ALA, CAN, K); confluence of Coppermine and Kendall rivers (NW side of Coppermine River, S side of Kendall River), 67°6′51.1″N, 116°8′18.3″W ± 150 m, 220 m, 7 July 2014, Saarela, Sokoloff & Bull 3581 (CAN); SE-facing slopes above Escape Rapids, W side of Coppermine River, 67°36′49.8″N, 115°29′27.4″W ± 10 m, 67 m, 8 July 2014, Saarela, Sokoloff & Bull 3739 (ALA, ALTA, CAN, MO, MT, O, UBC, US, WIN); Kugluk (Bloody Falls) Territorial Park, sandy rocky beach at W side of Bloody Falls, 67°44′27.8″N, 115°22′20.3″W ± 25 m, 5 m, 13 July 2014, Saarela, Sokoloff & Bull 3914 (ALA, ALTA, CAN); Kugluk (Bloody Falls) Territorial Park, rocky beach above Bloody Falls, W bank of Coppermine River, 67°44′18″N, 115°22′57.3″W ± 250 m, 34 m, 14 July 2014, Saarela, Sokoloff & Bull 3972 (ALA, CAN, K, MO, NY, QFA, UBC, US); Kugluk (Bloody Falls) Territorial Park, rocky sandy beach just below Bloody Falls, W side of Coppermine River, vicinity of confluence with small creek, beach seasonally flooded, 67°44′54.5″N, 115°22′17.2″W ± 75 m, 9 m, 17 July 2014, Saarela, Sokoloff & Bull 4119 (CAN, MO, O); Kugluk (Bloody Falls) Territorial Park, rocky sandy beach just below Bloody Falls, W side of Coppermine River, vicinity of confluence with small creek, beach seasonally flooded, 67°44′54.5″N, 115°22′17.2″W ± 75 m, 9 m, 17 July 2014, Saarela, Sokoloff & Bull 4122 (CAN, MT, WIN); clay slopes and beach on E side of Coppermine River, just above start of Bloody Falls, 67°44′9.4″N, 115°22′41.2″W ± 15 m, 40 m, 19 July 2014, Saarela, Sokoloff & Bull 4217 (ALA, ALTA, CAN, UBC).

Deschampsia sukatschewii subsp. borealis (Trautv.) Tzvelev—Hairgrass | Circumpolar | Noteworthy Collection

Newly recorded for the study area and western mainland Nunavut, closing a large distribution gap between Tuktut Nogait National Park and southeastern mainland Nunavut (Porsild & Cody, 1980; Barkworth, 2007; Saarela et al., 2013a). Our collection was gathered along the disturbed slopes of the Coppermine River upstream from Sandstone Rapids, growing with Arnica angustifolia, Betula glandulosa, Bromus pumpellianus, Dasiphora fruticosa, Equisetum arvense subsp. alpestre, Hedysarum americanum and Salix spp. Barkworth (2007), the treatment we used to key our Deschampsia P. Beauv. collections, did not recognise infraspecific taxa in Deschampsia sukatschewii. Elven et al. (2011) recognised three subspecies, including two recorded in Canada, subsp. borealis and subsp. orientalis (Hultén) Tzvelev. Chiapella et al. (2011) suggested the name Deschampsia sukatschewii is misapplied in the North American Arctic. Some authors have recognised the taxon as Deschampsia pumila (Griseb.) Ostenf. (Hultén, 1968; Porsild & Cody, 1980), an illegitimate homonym. According to the map in Aiken et al. (2007) this species is recorded from Banks, Baffin, Devon, Ellesmere, Prince Charles and Prince Patrick islands, and a few mainland sites.

Specimens Examined: Canada. Nunavut: Kitikmeot Region: slopes on E side of Coppermine River, N of its confluence with Fockler Creek, ca. 0.8 km SW of Sandstone Rapids, 67°26′36.9″N, 115°38′50.1″W ± 50 m, 128 m, 4 July 2014, Saarela, Sokoloff & Bull 3392 (ALA, CAN, K, O, QFA, UBC).

Dupontia fisheri R. Br.—Fisher’s tundra grass | Circumpolar

Previously recorded from Kugluktuk (Cody, 1954b), but this record is not mapped in Porsild & Cody (1980). We made collections at Richardson Bay, an island at the mouth of the Coppermine River, Kugluk (Bloody Falls) Territorial Park and Kugluktuk. Morphological and molecular evidence supports recognition of a single species in the Dupontia fisheri complex (Brysting et al., 2003, 2004), which is widespread throughout the Canadian Arctic Archipelago and recorded from several mainland sites (Porsild & Cody, 1980; Cody, Scotter & Zoltai, 1989; Cody, 1996a; Aiken et al., 2007; Saarela et al., 2013a). Elven et al. (2011) provisionally recognise three morphological variants that correspond to tetraploids, octoploids and dodecaploids, all of which are present in the Canadian Arctic.

Specimens Examined: Canada. Nunavut: Kitikmeot Region: Coppermine [Kugluktuk], near mission, 67°50′N, 115°7′W, 30 July 1951, W. I. Findlay 210 (DAO-172725, not seen); Coppermine [Kugluktuk], 67°49′36″N, 115°5′36″W, 31 July 1951, W. I. Findlay 215 (ACAD-30934, ALTA-VP-1781, DAO-172726, UBC-V40768); Kugluktuk, airport, 21 July 2013, 67.81749°N, 115.13449°W, B. A. Bennett 13-0633 (ALA, BABY, det. B. A. Bennett, 2013); Richardson Bay, confluence of Richardson and Rae rivers at Coronation Gulf, ca. 20 km WNW of Kugluktuk, 67°54′11.2″N, 115°32′27.4″W ± 200 m, 0 m, 8 July 2014, Saarela, Sokoloff & Bull 3688 (ALA, CAN, US); unnamed island just E (ca. 3.3 km) of Kugluktuk at mouth of Coppermine River, 67°49′29.2″N, 115°1′3.2″W ± 50 m, 1 m, 8 July 2014, Saarela, Sokoloff & Bull 3712 (CAN, UBC); Kugluk (Bloody Falls) Territorial Park, stream bed in deep gully, 0.75 km W of Bloody Falls, 67°44′40.2″N, 115°23′17.3″W ± 3 m, 46 m, 15 July 2014, Saarela, Sokoloff & Bull 4021 (ALTA, CAN, O); W of Kugluktuk on tundra flats above Coppermine River, S of 1 Coronation Drive and N of community power plant, 67°49′28.97″N, 115°5′0.2″W ± 100 m, 8 m, 22 July 2014, Saarela, Sokoloff & Bull 4265 (CAN, NY, QFA, UBC, US, WIN); flats below large overhanging cliffs above Coppermine River, just S of Kugluktuk, 67°48′56.7″N, 115°6′22.6″W ± 10 m, 2 m, 26 July 2014, Saarela, Sokoloff & Bull 4408 (CAN, MO, MT).

Elymus alaskanus (Scribn. & Merr.) Á. Löve subsp. alaskanus, Figs. 30B and 30C—Alaska wildrye | Amphi-Beringian (E) | Noteworthy Record

Newly recorded for the study area and western mainland Nunavut. A collection made by B. Bennett in 2013 at the Kugluktuk Airport was initially determined as Elymus violaceus, but better fits Elymus alaskanus (most glumes about half the length of the spikelet). We made collections at Fockler Creek, Coppermine Mountains and Kugluk (Bloody Falls) Territorial Park. The nearest collections to the west are from Tuktut Nogait National Park and vicinity (Saarela et al., 2013a) and to the east from the junction of Baillie and Back rivers (Tener 360 in 1955, CAN-235418, det. J. M. Saarela 2011), a collection not mapped in Porsild & Cody (1980) or Barkworth, Campbell & Saloman (2007). Taxonomy follows Barkworth, Campbell & Saloman (2007). Porsild & Cody (1980) included this taxon in their concept of Agropyron violaceum (Hornem.) Lange subsp. violaceum, treated here as Elymus violaceus. Aiken et al. (2007) did not recognise infraspecific taxa of Elymus alaskanus in the Canadian Arctic Archipelago.

Specimens Examined: Canada. Nunavut: Kitikmeot Region: Kugluktuk, airport, 21 July 2013, 67.81749°N, 115.13449°W, B. A. Bennett 13-0328 (ALA); spruce forest along Fockler Creek, ca. 2.3 km SSE of Sandstone Rapids, Coppermine River, 67°25′45.7″N, 115°37′21.8″W ± 25 m, 166 m, 2 July 2014, Saarela, Sokoloff & Bull 3191 (CAN, MO, MT, O); E side of Fockler Creek, in valley just above creek’s confluence with the Coppermine River, ca. 1.4 km SSW of Sandstone Rapids, 67°26′21.4″N, 115°38′54″W ± 5 m, 140 m, 4 July 2014, Saarela, Sokoloff & Bull 3357 (CAN, NY, QFA, WIN); Coppermine River, sandstone cliffs above Sandstone Rapids, 67°27′29.6″N, 115°37′59.3″W ± 100 m, 110 m, 6 July 2014, Saarela, Sokoloff & Bull 3473 (ALA, ALTA, CAN, O, UBC, US); flats atop and upper slopes of Coppermine Mountains, N/W side of Coppermine River, 67°14′53.6″N, 115°38′37.9″W ± 15 m, 401 m, 9 July 2014, Saarela, Sokoloff & Bull 3762 (CAN, UBC, US); Kugluk (Bloody Falls) Territorial Park, upper ledges of rocky (gabbro) S-facing cliffs above the start of Bloody Falls (W bank of River), just E of Portage Trail, 67°44′21.7″N, 115°22′42.2″W ± 25 m, 46 m, 14 July 2014, Saarela, Sokoloff & Bull 3937 (ALA, ALTA, CAN).

Elymus alaskanus subsp. hyperarcticus (Polunin) Á. Löve & D. Löve—Tundra wildrye | Asian (N/C)–amphi-Beringian–North American (N) | Noteworthy Record

Newly recorded for the study area, closing a distribution gap between sites on the Kent Peninsula (Hoare 1512, CAN-203806, det. J. M. Saarela 2011), southern Victoria Island (specimens at CAN) and eastern Great Slave Lake (Porsild & Porsild 4821, CAN-203085, det. J. M. Saarela 2011). We made collections at Big Creek and Kugluk (Bloody Falls) Territorial Park. Taxonomy follows Barkworth, Campbell & Saloman (2007). The taxon was treated as Agropyron violaceum var. hyperarcticum Polunin by Porsild & Cody (1980), who mapped only Agropyron violaceum s.l. In the Canadian Arctic Archipelago Aiken et al. (2007) recognised plants as Elymus alaskanus s.l.

Specimens Examined: Canada. Nunavut: Kitikmeot Region: forest and slopes at confluence of Big Creek and Coppermine River, N side of Coppermine River, S side of Coppermine Mountains, 67°14′29.3″N, 116°2′44.5″W ± 250 m, 180–199 m, 7 July 2014, Saarela, Sokoloff & Bull 3549 (CAN, MO, MT, O); Kugluk (Bloody Falls) Territorial Park, rocky valley immediately SW of Bloody Falls, along rough marked section of Portage Trail, head of small unnamed pond just W of falls, 67°44′42.8″N, 115°22′29.2″W ± 10 m, 9 m, 13 July 2014, Saarela, Sokoloff & Bull 3908 (CAN, UBC); Kugluk (Bloody Falls) Territorial Park, sandy rocky beach at W side of Bloody Falls, 67°44′27.8″N, 115°22′20.3″W ± 25 m, 5 m, 13 July 2014, Saarela, Sokoloff & Bull 3915 (CAN, NY, QFA, UBC, US, WIN); Kugluk (Bloody Falls) Territorial Park, rocky sandy beach just below Bloody Falls, W side of Coppermine River, vicinity of confluence with small creek, beach seasonally flooded, 67°44′54.5″N, 115°22′17.2″W ± 75 m, 9 m, 17 July 2014, Saarela, Sokoloff & Bull 4131 (ALA, ALTA, CAN).

Elymus violaceus (Hornem.) Feilberg, Fig. 31A—High wildyre | North American (N) | Noteworthy Record

Figure 31 Elymus violaceus and Festuca altaica.

Elymus violaceus: (A) habit, Saarela et al. 4138. Festuca altaica: (B) habit, Saarela et al. 3194. (C) Inflorescence, Saarela et al. 3194. Photographs by R. D. Bull.

First record for the study area. At a site in Kugluk (Bloody Falls) Territorial Park we found a single large cespitose plant growing on a gradual southeast-facing slope above a small creek running through a deep gully into the Coppermine River, growing with Equisetum arvense, Salix niphoclada and Symphyotrichum pygmaeum. At a site on the east side of the Coppermine River just before the start of Bloody Falls, outside the park boundary, it was locally common on the wet clay slopes and beach along the river, growing with Alopecurus borealis, Deschampsia cespitosa subsp. cespitosa, Equisetum arvense, Festuca rubra subsp. arctica, Hordeum jubatum subsp. intermedium, Juncus arcticus subsp. alaskanus and Poa pratensis subsp. alpigena. Recently recorded from a nearby site ca. 30 km west of the Coppermine River (Reading 97a, DAO; Cody & Reading, 2005). The next-nearest known sites are from Bathurst Inlet (Anderson s.n., CAN-203108, det. J. M. Saarela 2011), Hood River (Gould & Walker, 1997) and the McTavish Arm of Great Bear Lake (Shacklette 2876, CAN-200030, det. J. M. Saarela 2011). Taxonomy follows Barkworth, Campbell & Saloman (2007) and Elven et al. (2011), who recognise this taxon at species level. Harrison & Hebda (2011) suggested Elymus violaceus should be treated as a subspecies of Elymus alaskanus, with the correct name being Elymus alaskanus subsp. latiglumis (Scribn. & J. G. Sm.), while Campbell and Soreng in Soreng et al. (2003) treated it as Elymus trachycaulus subsp. violaceus (Hornem.) Á Löve & D. Löve. Aiken et al. (2007) included it in their concept of Elymus alaskanus s.l.

Specimens Examined: Canada. Nunavut: Kitikmeot Region: Kugluk (Bloody Falls) Territorial Park, W side of Coppermine River, below Bloody Falls, 67°45′7.3″N, 115°22′20.1″W ± 3 m, 12 m, 17 July 2014, Saarela, Sokoloff & Bull 4138 (ALA, ALTA, CAN, MO, O, UBC); clay slopes and beach on E side of Coppermine River, just above start of Bloody Falls, 67°44′9.4″N, 115°22′41.2″W ± 15 m, 40 m, 19 July 2014, Saarela, Sokoloff & Bull 4218 (CAN, US).

Festuca altaica Trin., Figs. 31B and 31C—Altai fescue, rough fescue | Amphi-Beringian & North American (NE) | Noteworthy Record

Newly recorded for Nunavut (Porsild & Cody, 1980; Pavlick & Looman, 1984; Aiken & Darbyshire, 1990; Darbyshire & Pavlick, 2007) and an eastern range extension from the nearest collections from Great Bear Lake (Porsild, 1943; Porsild & Cody, 1980). We did not encounter it further north than Kugluk (Bloody Falls) Territorial Park. At the four sites where we collected the species, it was common to locally common, growing in dense shrub tundra, open meadows and on disturbed slopes, with Arnica angustifolia, Betula glandulosa, Bromus pumpellianus, Equisetum arvense subsp. alpestre, Dasiphora fruticosa, Juniperus communis subsp. depressa, Lupinus arcticus, Pyrola grandiflora and Shepherdia canadensis. It also reaches the Canadian Low Arctic in Tuktut Nogait National Park and vicinity and northern Yukon (Porsild, 1943; Porsild & Cody, 1980; Saarela et al., 2013a), and is not known from the Canadian Arctic Archipelago.

Specimens Examined: Canada. Nunavut: Kitikmeot Region: spruce forest along Fockler Creek, ca. 2.3 km SSE of Sandstone Rapids, Coppermine River, 67°25′45.7″N, 115°37′21.8″W ± 25 m, 166 m, 2 July 2014, Saarela, Sokoloff & Bull 3194 (ALA, ALTA, CAN, K, NY, QFA, UBC); slopes on E side of Coppermine River, N of its confluence with Fockler Creek, ca. 0.8 km SW of Sandstone Rapids, 67°26′36.9″N, 115°38′50.1″W ± 50 m, 128 m, 4 July 2014, Saarela, Sokoloff & Bull 3384 (CAN, K, MO, MT, NY, O, QFA, US); confluence of Coppermine River and Melville Creek, just W of Coppermine Mountains, 67°15′52″N, 115°30′55.3″W ± 350 m, 178–190 m, 7 July 2014, Saarela, Sokoloff & Bull 3500 (ALTA, CAN, UBC, US); Kugluk (Bloody Falls) Territorial Park, rocky valley immediately SW of Bloody Falls, along rough marked section of Portage Trail, 67°44′34″N, 115°22′16″W ± 50 m, 20 m, 13 July 2014, Saarela, Sokoloff & Bull 3910 (ALA, CAN, MO, MT, O, WIN).

Festuca baffinensis Polunin—Baffin Island fescue | Asian (NE)–Amphi-Beringian–North American–amphi-Atlantic

We did not find this species in 2014, but it has been mapped for the study area in two publications (Porsild & Cody, 1980; Aiken & Darbyshire, 1990). The only early collection we located was taken at Rae River, and we assume this is the record mapped for the region. The specimen bears the following handwritten note: “This came from the Brit. Museum with a lot of duplicates of Richardson. It was labelled Rae River, but Richardson was never there so that it was probably collected by Parry or some other early explorer. Rae River is N.W. from Southampton Island”. Bruce Bennett collected it at the Kugluktuk Airport in 2013. Widespread throughout the Canadian Arctic Archipelago and known from several mainland Arctic sites (Porsild & Cody, 1980; Aiken et al., 2007; Saarela et al., 2013a).

Specimens Examined: Canada. Nunavut: Kitikmeot Region: Kugluktuk, airport, 21 July 2013, 67.81749°N, 115.13449°W, B. A. Bennett 13-0631 (CAN); Rae River, s.d., s.c. (CAN-503878).

Festuca brachyphylla Schult. & Schult. f. subsp. brachyphylla—Alpine fescue | Circumpolar–alpine

Reported for the study area in Porsild & Cody (1980), but not in Aiken & Darbyshire (1990), in which the nearest mapped record is from Bernard Harbour. We were unable to locate a voucher specimen. We made collections at Kugluktuk, Fockler Creek, Bigtree River, Kugluk (Bloody Falls) Territorial Park and Heart Lake. Widespread throughout the Canadian Arctic Archipelago and known from several mainland Arctic sites (Porsild & Cody, 1980; Cody, Scotter & Zoltai, 1989; Korol, 1992; Cody & Reading, 2005; Aiken et al., 2007; Saarela et al., 2013a).

Specimens Examined: Canada. Nunavut: Kitikmeot Region: Kugluktuk, rocky slopes of North Hill, 67°49′31.4″N, 115°6′54″W ± 100 m, 42 m, 29 June 2014, Saarela, Sokoloff & Bull 3077 (CAN, UBC); flats on W side of Fockler Creek, above spruce forest in creek valley, ca. 2.2 km S of Sandstone Rapids, Coppermine River, 67°25′49″N, 115°37′55″W ± 50 m, 152 m, 1 July 2014, Saarela, Sokoloff & Bull 3117 (ALA, CAN); old riverbed of Fockler Creek, ca. 2.3 km SSE of Sandstone Rapids, Coppermine River, 67°25′45.7″N, 115°37′21.8″W ± 25 m, 166 m, 2 July 2014, Saarela, Sokoloff & Bull 3182 (ALA, ALTA, CAN, MO, MT, O, QFA, UBC, US, WIN); meadow just S of Tundra Lake, ca. 4.2 km SE of Sandstone Rapids, Coppermine River, 67°25′34.8″N, 115°33′27.8″W ± 20 m, 265 m, 5 July 2014, Saarela, Sokoloff & Bull 3436 (CAN); confluence of Coppermine and Bigtree rivers, 66°56′23.8″N, 116°21′3.2″W ± 100 m, 265 m, 7 July 2014, Saarela, Sokoloff & Bull 3605 (ALTA, CAN); flats atop and upper slopes of Coppermine Mountains, N/W side of Coppermine River, 67°14′43.7″N, 115°38′51.2″W ± 150 m, 422 m, 9 July 2014, Saarela, Sokoloff & Bull 3752 (CAN, O); Kugluk (Bloody Falls) Territorial Park, rocky cliffs and ledges directly above (W side) of Bloody Falls, just S of heavily used day-use/fishing area, 67°44′40.1″N, 115°22′4.9″W ± 20 m, 8 m, 12 July 2014, Saarela, Sokoloff & Bull 3804 (CAN, MO, MT); Heart Lake, SW of Kugluktuk, 6.4 km SW of mouth of Coppermine River, 67°48′6.7″N, 115°13′40.6″W ± 50 m, 41 m, 23 July 2014, Saarela, Sokoloff & Bull 4300 (CAN, UBC, US).

Festuca rubra subsp. arctica (Hack.) Govor.—Richardson’s red fescue | Circumpolar

Previously recorded for the study area in Aiken & Darbyshire (1990), but not in Porsild & Cody (1980), in which the nearest mapped record is from Bernard Harbour. We were unable to locate any earlier collections from the study area. We made collections at Fockler Creek, Sandstone Rapids, Melville Creek and Kugluk (Bloody Falls) Territorial Park. Taxonomy follows Darbyshire & Pavlick (2007). Porsild & Cody (1980) and Elven et al. (2011) treated the taxon as Festuca rubra subsp. richardsonii (Hook.) Hultén, the latter noting arctica may be the correct name at subspecific rank but that the type of the basionym, Festuca rubra f. arctica Hack., is unknown. Aiken & Darbyshire (1990) treated it as Festuca richardsonii Hook. Elsewhere in the Canadian Arctic known from the western Arctic islands (Banks, Melville and Victoria islands) as well as mainland sites (Yukon, Northwest Territories, Nunavut, northern Quebec) (Porsild & Cody, 1980; Aiken & Darbyshire, 1990; Cody & Reading, 2005; Saarela et al., 2013a; Bennett, 2015).

Specimens Examined: Canada. Nunavut: Kitikmeot Region: Kugluktuk Airport, near weather station, 67°49′1.3″N, 115°8′2.3″W ± 25 m, 27 m, 30 June 2014, Saarela, Sokoloff & Bull 3107 (CAN, UBC, US); slopes on E side of Coppermine River, N of its confluence with Fockler Creek, ca. 0.8 km SW of Sandstone Rapids, 67°26′36.9″N, 115°38′50.1″W ± 50 m, 128 m, 4 July 2014, Saarela, Sokoloff & Bull 3382 (CAN, MT, UBC, US); Coppermine River, sandstone cliffs above Sandstone Rapids, 67°27′29.6″N, 115°37′59.3″W ± 100 m, 110 m, 6 July 2014, Saarela, Sokoloff & Bull 3477 (CAN, NY, QFA, WIN); confluence of Coppermine River and Melville Creek, just W of Coppermine Mountains, 67°15′52″N, 115°30′55.3″W ± 350 m, 178–190 m, 7 July 2014, Saarela, Sokoloff & Bull 3510 (ALA, ALTA, CAN, MO, O); Kugluk (Bloody Falls) Territorial Park, rocky cliffs and ledges directly above (W side) of Bloody Falls, just S of heavily used day-use/fishing area, 67°44′40.1″N, 115°22′4.9″W ± 20 m, 8 m, 12 July 2014, Saarela, Sokoloff & Bull 3801 (ALA, ALTA, CAN); Kugluk (Bloody Falls) Territorial Park, rocky cliffs and ledges directly above (W side) of Bloody Falls, just S of heavily used day-use/fishing area, 67°44′40.1″N, 115°22′4.9″W ± 20 m, 8 m, 12 July 2014, Saarela, Sokoloff & Bull 3798b (CAN, MO, O); Kugluk (Bloody Falls) Territorial Park, terrace above S-facing slopes above start of Bloody Falls, W side of Coppermine River, 67°44′27.2″N, 115°22′58″W ± 50 m, 68 m, 16 July 2014, Saarela, Sokoloff & Bull 4092 (CAN, MT, UBC, US, WIN).

Festuca rubra L. subsp. rubra—Red fescue | Circumboreal-polar | Noteworthy Record

Newly recorded for the study area, closing a large distribution gap between Bathurst Inlet, Hood River, Great Slave Lake and the Mackenzie Delta area (Porsild & Cody, 1980; Aiken & Darbyshire, 1990; Gould & Walker, 1997). Porsild & Cody’s (1980) map (for Festuca rubra s.l.) does not distinguish between subsp. rubra and subsp. arctica (as subsp. richardsonii), although the same distribution gap is largely present, with the exception of a record along the coast intermediate between these sites that is probably subsp. arctica. Two of our collections (nos. 4398 and 4392) were large vigorous plants (e.g., no. 4398 approximately 70 cm tall) gathered in Kugluktuk in highly disturbed areas where they may have been planted, while no. 3798a was collected in a natural but heavily human-used area: the rocky cliffs and ledges directly above (west side) of Bloody Falls, growing with Anthoxanthum monticola subsp. alpinum, Arnica angustifolia, Calamagrostis purpurascens, Dryopteris fragrans, Poa glauca subsp. glauca, Potentilla spp. and Saxifraga tricuspidata. The subspecies rubra differs from subsp. arctica by the vestiture of the spikelets, which are glabrous to scabrous (scabridulous in our plants) in subsp. rubra and densely pubescent in subsp. arctica. This subspecies is known from a few other Canadian Arctic sites, including Baffin Island (Iqaluit and Clyde River), Eglinton Island (needs confirmation) and a few scattered mainland sites (Aiken & Darbyshire, 1990; Aiken et al., 2007). Other Arctic communities should be searched for this introduced subspecies.

Specimens Examined: Canada. Nunavut: Kitikmeot Region: Kugluk (Bloody Falls) Territorial Park, rocky cliffs and ledges directly above (W side) of Bloody Falls, just S of heavily used day-use/fishing area, 67°44′40.1″N, 115°22′4.9″W ± 20 m, 8 m, 12 July 2014, Saarela, Sokoloff & Bull 3798a (CAN, UBC, US); grassy yard in Kugluktuk, 67°49′27.4″N, 115°5′26.2″W ± 3 m, 29 m, 26 July 2014, Saarela, Sokoloff & Bull 4392 (ALA, ALTA, CAN, MO, MT, O); Kugluktuk, ball diamond on S side of town, 67°49′23.2″N, 115°6′31.9″W ± 5 m, 29 m, 26 July 2014, Saarela, Sokoloff & Bull 4398 (CAN, K, NY, QFA, UBC, US, WIN).

Festuca viviparoidea Krajina ex Pavlick subsp. viviparoidea, Fig. 32A—Viviparous fescue | Amphi-Atlantic–European (N)–Asian (NW) & amphi-Beringian | Noteworthy Record

Figure 32 Festuca viviparoidea and Hordeum jubatum subsp. intermedium.

Festuca viviparoidea: (A) habit, Saarela et al. 3938. Hordeum jubatum subsp. intermedium. (B) Habit, Saarela et al. 4213. (C) Habitat, Saarela et al. 4213. Photographs by J. M. Saarela (A) and R. D. Bull (B, C).

Newly recorded for the study area and mainland Nunavut, and an eastern range extension from the nearest known sites in Tuktut Nogait National Park and vicinity (Saarela et al., 2013a). We found this taxon in Kugluk (Bloody Falls) Territorial Park growing on rocky ledges above the start of Bloody Falls with Anthoxanthum monticola subsp. alpinum, Arnica angustifolia, Calamagrostis purpurascens, Dryopteris fragrans, Poa glauca subsp. glauca, Potentilla arenosa subsp. arenosa, Potentilla nivea and Saxifraga tricuspidata, adjacent to our collection site of Allium schoenoprasum. Elsewhere in Nunavut known from northern Ellesmere Island (Aiken & Darbyshire, 1990; Aiken et al., 2007) and the Belcher Islands (Consaul et al. 4045, 4023, 4112, CAN). In Northwest Territories known from the southernmost tip of Banks Island (Gillespie et al. 7204, 7239, CAN), and also recorded from Arctic Yukon, as Festuca vivipara subsp. glabra Fred. (Aiken & Darbyshire, 1990). The name Festuca vivipara (L.) Sm. has been misapplied in North America. Our collection matches the descriptions in Aiken & Darbyshire (1990: 77) and Darbyshire & Pavlick (2007), with the exception of the culms immediately below the inflorescence, which are glabrous, not sparsely to densely pubescent as described for the taxon. The other subspecies, Festuca viviparoidea subsp. krajinae Pavlick, has a more southerly distribution in alpine areas of western Canada and Alaska (Darbyshire & Pavlick, 2007).

Specimens Examined: Canada. Nunavut: Kitikmeot Region: Kugluk (Bloody Falls) Territorial Park, upper ledges of rocky (gabbro) S-facing cliffs above the start of Bloody Falls (W bank of River), just E of Portage Trail, 67°44′21.7″N, 115°22′42.2″W ± 25 m, 46 m, 14 July 2014, Saarela, Sokoloff & Bull 3938 (ALA, CAN, UBC).

Hordeum jubatum subsp. intermedium Bowden, Figs. 32B and 32C—Intermediate barley | North America | Noteworthy Record

Newly recorded for the for the study area. Our collections fill a distribution gap between Bathurst Inlet, upper Hood River and Great Bear Lake (Porsild & Cody, 1980; Gould & Walker, 1997). We encountered a large population in wet clay substrate on a slope running into and along the river bank just above Bloody Falls on the east side of the Coppermine River, where the species appears to be native, growing with Alopecurus magellanicus, Juncus arcticus subsp. alaskanus, Carex aquatilis subsp. stans, Deschampsia cespitosa, Equisetum arvense subsp. alpestre, Festuca rubra subsp. arctica, and Poa pratensis subsp. alpigena. At beach sites below Bloody Falls, within the park, we observed only a few plants. These likely originated from the large population upstream on the other side of the river. It was also growing sporadically throughout Kugluktuk along disturbed roadsides, where it may have been introduced unintentionally, with Leymus mollis subsp. villosissimus and Taraxacum spp. The distribution and taxonomy of Hordeum jubatum in the Canadian Arctic is reviewed in Gillespie et al. (2015), though they did not include the record from Hood River (Gould & Walker, 1997). Elven et al. (2011) recognised only Hordeum jubatum L. subsp. jubatum as part of the Panarctic flora. In the Canadian Arctic subsp. intermedium is known from Bathurst Inlet and southern Baffin Island. The Hood River record(s) are only determined to species in Gould & Walker (1997). On mainland Nunavut also recorded from the Nueltin Lake area where it was found growing near a building (Porsild, 1950b; Porsild & Cody, 1980).

Specimens Examined: Canada. Nunavut: Kitikmeot Region: Kugluk (Bloody Falls) Territorial Park, rocky beach above Bloody Falls, W bank of Coppermine River, 67°44′18″N, 115°22′57.3″W ± 250 m, 34 m, 14 July 2014, Saarela, Sokoloff & Bull 3973 (CAN); Kugluk (Bloody Falls) Territorial Park, rocky sandy beach just below Bloody Falls, W side of Coppermine River, vicinity of confluence with small creek, beach seasonally flooded, 67°44′54.5″N, 115°22′17.2″W ± 75 m, 9 m, 17 July 2014, Saarela, Sokoloff & Bull 4133 (CAN); clay slopes and beach on E side of Coppermine River, just above start of Bloody Falls, 67°44′9.4″N, 115°22′41.2″W ± 15 m, 40 m, 19 July 2014, Saarela, Sokoloff & Bull 4213 (ALA, ALTA, CAN, MO, MT, O, UBC); roadside and grassy area around buildings in Kugluktuk, 67°49′34″N, 115°5′27.1″W ± 20 m, 16 m, 24 July 2014, Saarela, Sokoloff & Bull 4347 (CAN, UBC, US).

Leymus mollis subsp. villosissimus (Scribn.) Á. Löve & D. Löve, Fig. 33—Sea lyme-grass, American dune grass | Asian (NE)–amphi-Beringian–North American (N)

Figure 33 Leymus mollis subsp. villosissimus.

(A) Habitat, Saarela et al. 4174. (B) Habit, Kugluktuk, Nunavut, 27 July 2014. Photographs J. M. Saarela (A) and P. C. Sokoloff (B).

Previously recorded from Kugluktuk (Cody, 1954b; Bowden, 1957; Porsild & Cody, 1980; Barkworth & Atkins, 1984). All but one of our collections were made along the coast at Richardson Bay, Kugluktuk and an island in the mouth of the Coppermine River. We found a single, large population inland at the base of steep sandy banks along a braided stream in a deep gully just west of the Kugluk (Bloody Falls) Territorial Park boundary. Gould & Walker (1997) also recorded this species (but the other subspecies, as Elymus arenarius subsp. mollis (Trin.) Hultén) at inland sites along the Hood River. The taxon was previously recognised in the genus Elymus L. (Bowden, 1957; Porsild & Cody, 1980). The collection Findlay 244 at DAO is intermediate between this subspecies and Leymus mollis (Trin.) Pilg. subsp. mollis (duplicates at ACAD, ALTA and UBC not seen). Intermediates from elsewhere are reported in Bowden (1957). Elsewhere in the Canadian Arctic this subspecies is recorded from Banks, Baffin, King William, Southampton and Victoria islands, and numerous mainland sites (Aiken et al., 2007; Saarela et al., 2013a).

Specimens Examined: Canada. Nunavut: Kitikmeot Region: Coppermine [Kugluktuk] [67°49′36″N, 115°5′36″W ± 1.5 km], 25 August 1934, A. H. Dutilly 233 (36) (CAN-514252); Coppermine River, Fort Hearne–Bloody Falls [67.7761972°N, 115.2037222°W ± 7.5 km], 1931, A. M. Berry 1 (CAN-40050); Coppermine [Kugluktuk], 67°49′36″N, 115°5′36″W, 3 August 1951, W. I. Findlay 244 (ACAD-30936, ALTA-VP-1868, DAO-172957 01-01000172957, UBC-V40770); Richardson Bay, confluence of Richardson and Rae rivers at Coronation Gulf, ca. 20 km WNW of Kugluktuk, 67°54′11.2″N, 115°32′27.4″W ± 200 m, 0 m, 8 July 2014, Saarela, Sokoloff & Bull 3677 (CAN, MO, MT, US); unnamed island just E (ca. 3.3 km) of Kugluktuk at mouth of Coppermine River, 67°49′29.2″N, 115°1′3.2″W ± 50 m, 1 m, 8 July 2014, Saarela, Sokoloff & Bull 3710 (ALA, CAN); deep gully in sand hills, NW of Bloody Falls, 67°45′22.8″N, 115°22′56.9″W ± 3 m, 42 m, 18 July 2014, Saarela, Sokoloff & Bull 4174 (CAN, NY, QFA, WIN); W of Kugluktuk on tundra flats above Coppermine River, S of 1 Coronation Drive and N of community power plant, 67°49′28.97″N, 115°5′0.2″W ± 100 m, 8 m, 22 July 2014, Saarela, Sokoloff & Bull 4253 (ALTA, CAN, O); Heart Lake, SW of Kugluktuk, 6.4 km SW of mouth of Coppermine River, 67°48′6.7″N, 115°13′40.6″W ± 50 m, 41 m, 23 July 2014, Saarela, Sokoloff & Bull 4303 (CAN, K, UBC, US).

Phippsia algida (Sol.) R. Br.—Icegrass | Circumpolar | Noteworthy Record

Our collection, from Kugluk (Bloody Falls) Territorial Park, is the first record for the study area and the central mainland Arctic. Elsewhere on mainland Nunavut there are three more easterly collections, and the nearest mainland sites to the west are from the Mackenzie Delta area (Porsild & Cody, 1980; Cody, Scotter & Zoltai, 1989; Consaul & Aiken, 2007). The plants were uncommon and grew on a sandy, rocky beach with Artemisia tilesii, Chamerion latifolium, Equisetum arvense subsp. alpestre and Salix planifolia. Widespread throughout the Canadian Arctic Archipelago and recorded for northern Quebec and Labrador (Aiken et al., 2007).

Specimens Examined: Canada. Nunavut: Kitikmeot Region: Kugluk (Bloody Falls) Territorial Park, rocky sandy beach just below Bloody Falls, W side of Coppermine River, vicinity of confluence with small creek, beach seasonally flooded, 67°44′54.5″N, 115°22′17.2″W ± 75 m, 9 m, 17 July 2014, Saarela, Sokoloff & Bull 4123 (CAN, UBC).

Poa alpina L., Figs. 34A and 34B—Alpine bluegrass | Amphi-Beringian–North American–amphi-Atlantic–European–Asian (NW-C) | Noteworthy Record

Figure 34 Poa alpina and Poa glauca subsp. glauca.

Poa alpina: (A) inflorescence, Saarela et al. 3385. (B) Habit, Saarela et al. 3385. Poa glauca subsp. glauca: (C) habit, Saarela et al. 3631. (D) Habitat, Saarela et al. 3114. Photographs by R. D. Bull (A, B), P. C. Sokoloff (C), and J. M. Saarela (D).

Newly recorded for the study area. We made collections at Fockler Creek, Sandstone Rapids, Melville Creek, Heart Lake and Kugluktuk. There are scattered collections from mainland Nunavut and a few from the western Canadian Arctic Archipelago (Porsild & Cody, 1980; Cody, Scotter & Zoltai, 1984; Korol, 1992; Gould & Walker, 1997; Cody & Reading, 2005; Aiken et al., 2007; Soreng, 2007; Bennett, 2015). Recently recorded from September (Mouse) Lake, east of the Coppermine River (Reading 206, DAO; Cody & Reading, 2005). This specimen was published as no. 118, but the specimen label bears no. 206, with 118 scratched out. These collections fill in a conspicuous gap along the northern edge of the species’ mainland distribution between Great Bear Lake and Bathurst Inlet (Cody, Scotter & Zoltai, 1984). Elsewhere in the Canadian Arctic recorded from Baffin, Coats and Southampton islands, and adjacent northern Quebec and Labrador (Aiken et al., 2007).

Specimens Examined: Canada. Nunavut: Kitikmeot Region: E side of Fockler Creek, just above its confluence with Coppermine River, ca. 1.1 km SW of Sandstone Rapids, 67°26′30.6″N, 115°39′4.3″W ± 50 m, 135 m, 4 July 2014, Saarela, Sokoloff & Bull 3370 (CAN, UBC); slopes on E side of Coppermine River, N of its confluence with Fockler Creek, ca. 0.8 km SW of Sandstone Rapids, 67°26′36.9″N, 115°38′50.1″W ± 50 m, 128 m, 4 July 2014, Saarela, Sokoloff & Bull 3385 (CAN, MO, O); Coppermine River, sandstone cliffs above Sandstone Rapids, 67°27′29.6″N, 115°37′59.3″W ± 100 m, 110 m, 6 July 2014, Saarela, Sokoloff & Bull 3475 (CAN, US); confluence of Coppermine River and Melville Creek, just W of Coppermine Mountains, 67°15′52″N, 115°30′55.3″W ± 350 m, 178–190 m, 7 July 2014, Saarela, Sokoloff & Bull 3513 (ALA, CAN); S-facing sandstone cliffs above Coppermine River, ca. 7.8 km NNE of Sandstone Rapids, 67°31′15.1″N, 115°36′19.1″W ± 50 m, 106 m, 8 July 2014, Saarela, Sokoloff & Bull 3632 (CAN, MT, UBC); Heart Lake, SW of Kugluktuk, 6.4 km SW of mouth of Coppermine River, 67°48′7.8″N, 115°13′22.7″W ± 350 m, 33 m, 23 July 2014, Saarela, Sokoloff & Bull 4296 (CAN, US, WIN); Kugluktuk, ball diamond on S side of town, 67°49′23.2″N, 115°6′31.9″W ± 5 m, 29 m, 26 July 2014, Saarela, Sokoloff & Bull 4400 (ALTA, CAN).

Poa arctica R. Br. subsp. arctica—Arctic bluegrass | Circumpolar–alpine

Reported for the study area in Porsild & Cody (1980), but we were unable to locate a voucher specimen for confirmation. We made collections at Fockler Creek and Kugluk (Bloody Falls) Territorial Park. Taxonomy follows Soreng (2007), who recognised five subspecies in North America, of which three occur in the Arctic: subspp. arctica, caespitans Simmons ex Nannf. and lanata (Scribn. & Merr.) Soreng. These same taxa are recognised in Elven et al. (2011). Only subsp. arctica occurs in the study area. It is widespread throughout the Canadian Arctic (Porsild & Cody, 1980; Cody, Scotter & Zoltai, 1984, 1989; Korol, 1992; Aiken et al., 2007; Saarela et al., 2013a).

Specimens Examined: Canada. Nunavut: Kitikmeot Region: flats on W side of Fockler Creek, above spruce forest in creek valley, ca. 2.2 km S of Sandstone Rapids, Coppermine River, 67°25′49″N, 115°37′55″W ± 50 m, 152 m, 1 July 2014, Saarela, Sokoloff & Bull 3116 (CAN); NW-facing slope just upstream of small tributary from its confluence with Fockler Creek, ca. 2.4 km SSW of Sandstone Rapids, Coppermine River, 67°25′46″N, 115°38′49.4″W ± 200 m, 149 m, 3 July 2014, Saarela, Sokoloff & Bull 3307 (CAN, UBC); S of Fockler Creek, S-facing slope on N side of small tributary of Fockler Creek, ca. 2.3 km S of Sandstone Rapids, Coppermine River, 67°25′46.3″N, 115°38′2.5″W ± 5 m, 156 m, 6 July 2014, Saarela, Sokoloff & Bull 3459 (CAN); Kugluk (Bloody Falls) Territorial Park, flats above boardwalk W of Bloody Falls, 67°44′34.5″N, 115°22′27″W ± 100 m, 135 m, 16 July 2014, Saarela, Sokoloff & Bull 4060 (ALA, ALTA, CAN, US).

Poa glauca Vahl subsp. glauca, Figs. 34C and 34D—Glaucus bluegrass | Circumpolar–alpine

Previously reported for the study area (Porsild & Cody, 1980), but we were unable to locate a voucher specimen. We made numerous collections, including at Kugluktuk, Fockler Creek and Kugluk (Bloody Falls) Territorial Park. Taxonomy follows Soreng (2007), who recognised three infraspecific taxa in North America, of which two reach the Arctic: the widespread subsp. glauca (Porsild & Cody, 1980; Cody, Scotter & Zoltai, 1989; Korol, 1992; Cody & Reading, 2005; Aiken et al., 2007; Saarela et al., 2013a) and the viviparous, amphi-Beringian var. pekulnejensis (Jurtzev & Tzvelev) Prob. known in North America only from the Seward Peninsula, Alaska (Elven et al., 2011). Elven et al. (2011) recognised the latter taxon at species level, Poa pekulnejensis Jurtzev & Tzvelev.

Specimens Examined: Canada. Nunavut: Kitikmeot Region: Kugluktuk, airport, 67.816667°N, 115.143889°W, 2 July 2008, L. J. Gillespie, J. M. Saarela, L. M. Consaul & R. D. Bull 7457-1 (CAN-592321) & 7457-2 (CAN-592320); Kugluktuk, airport, 21 July 2013, 67.81749°N, 115.13449°W, B. A. Bennett 13-0632 (UBC; CAN; BABY; ALA, det. B. A. Bennett, December 2013); Kugluktuk, rocky slopes of North Hill, 67°49′31.4″N, 115°6′54″W ± 100 m, 42 m, 29 June 2014, Saarela, Sokoloff & Bull 3076 (ALTA, CAN, US); flats on W side of Fockler Creek, above spruce forest in creek valley, ca. 2.2 km S of Sandstone Rapids, Coppermine River, 67°25′49″N, 115°37′55″W ± 50 m, 152 m, 1 July 2014, Saarela, Sokoloff & Bull 3114 (CAN, NY, QFA, WIN); second ridge N of Fockler Creek, ca. 1.9 km SSE of Sandstone Rapids, Coppermine River, 67°26′2.4″N, 115°37′26.5″W ± 25 m, 187 m, 2 July 2014, Saarela, Sokoloff & Bull 3217 (ALA, CAN, O); SW-facing slope above (N side) of Fockler Creek, ca. 3.2 km SE of Sandstone Rapids, Coppermine River, 67°25′26.2″N, 115°36′14″W ± 25 m, 193 m, 5 July 2014, Saarela, Sokoloff & Bull 3415 (ALA, ALTA, CAN, K); S-facing slopes on W side of Coppermine River, about halfway between Escape Rapids and Muskox Rapids, 67°31′18.2″N, 115°36′20.1″W ± 150 m, 115 m, 8 July 2014, Saarela, Sokoloff & Bull 3631 (CAN, MO, MT); S-facing sandstone cliffs above Coppermine River, ca. 7.8 km NNE of Sandstone Rapids, 67°31′15.1″N, 115°36′19.1″W ± 50 m, 106 m, 8 July 2014, Saarela, Sokoloff & Bull 3633 (CAN, UBC); S-facing slopes above Coppermine River and below spruce forest, ca. 7.8 km NNE of Sandstone Rapids, 67°31′16.2″N, 115°36′52.1″W ± 200 m, 110 m, 8 July 2014, Saarela, Sokoloff & Bull 3658 (ALA, CAN, K, NY, QFA); flats atop and upper slopes of Coppermine Mountains, N/W side of Coppermine River, 67°14′49.9″N, 115°38′43.7″W ± 200 m, 467 m, 9 July 2014, Saarela, Sokoloff & Bull 3778 (CAN, K, NY, QFA, UBC, US, WIN); Kugluk (Bloody Falls) Territorial Park, rocky cliffs and ledges directly above (W side) of Bloody Falls, just S of heavily used day-use/fishing area, 67°44′40.1″N, 115°22′4.9″W ± 20 m, 8 m, 12 July 2014, Saarela, Sokoloff & Bull 3802 (CAN, MO, MT, O); Kugluk (Bloody Falls) Territorial Park, rocky valley immediately SW of Bloody Falls, along rough marked section of Portage Trail, 67°44′34″N, 115°22′16″W ± 50 m, 20 m, 13 July 2014, Saarela, Sokoloff & Bull 3927 (CAN, UBC, US); W of Kugluktuk on tundra flats above Coppermine River, S of 1 Coronation Drive and N of community power plant, 67°49′28.97″N, 115°5′0.2″W ± 100 m, 8 m, 22 July 2014, Saarela, Sokoloff & Bull 4259 (ALTA, CAN, MO, MT, O); W of Kugluktuk on tundra flats above Coppermine River, S of 1 Coronation Drive and N of power plant, 67°49′28.97″N, 115°5′0.2″W ± 100 m, 8 m, 25 July 2014, Saarela, Sokoloff & Bull 4387 (CAN, UBC, US, WIN).

Poa pratensis subsp. alpigena (Lindm.) Hiitonen—Northern meadow-grass | Circumboreal-polar | Noteworthy Record

Newly recorded for the study area, closing a distribution gap between eastern Great Bear Lake and the Bathurst Inlet area (Porsild & Cody, 1980; Cody, Scotter & Zoltai, 1984; Cody & Reading, 2005). We made collections at Fockler Creek, Coppermine Mountains, Kugluk (Bloody Falls) Territorial Park and Kugluktuk. Although widespread throughout the Canadian Arctic Archipelago and the circumboreal region, there are comparatively few collections from mainland Nunavut (Porsild & Cody, 1980; Cody, Scotter & Zoltai, 1984, 1989; Korol, 1992; Gould & Walker, 1997; Cody & Reading, 2005; Soreng, 2007; Saarela et al., 2013a).

Specimens Examined: Canada. Nunavut: Kitikmeot Region: Kugluktuk, corner of Tuktu Road and Saddleback Street, 67°49′23.8″N, 115°6′17.6″W ± 10 m, 21 m, 29 June 2014, Saarela, Sokoloff & Bull 3092 (CAN, UBC); spruce forest along Fockler Creek, ca. 2.3 km SSE of Sandstone Rapids, Coppermine River, 67°25′45.7″N, 115°37′21.8″W ± 25 m, 166 m, 2 July 2014, Saarela, Sokoloff & Bull 3193 (CAN); E side of Fockler Creek, just above its confluence with Coppermine River, ca. 1.1 km SW of Sandstone Rapids, 67°26′30.6″N, 115°39′4.3″W ± 50 m, 135 m, 4 July 2014, Saarela, Sokoloff & Bull 3371b (ALTA, CAN); flats atop and upper slopes of Coppermine Mountains, N/W side of Coppermine River, 67°14′53.6″N, 115°38′37.9″W ± 15 m, 401 m, 9 July 2014, Saarela, Sokoloff & Bull 3760 (CAN, US); Kugluk (Bloody Falls) Territorial Park, rocky cliffs and ledges directly above (W side) of Bloody Falls, just S of heavily used day-use/fishing area, 67°44′40.1″N, 115°22′4.9″W ± 20 m, 8 m, 12 July 2014, Saarela, Sokoloff & Bull 3803 (CAN, MO, O); Kugluk (Bloody Falls) Territorial Park, rocky cliffs and ledges directly above (W side) of Bloody Falls, just S of heavily used day-use/fishing area, 67°44′40.1″N, 115°22′4.9″W ± 20 m, 8 m, 12 July 2014, Saarela, Sokoloff & Bull 3805 (CAN, NY, QFA, WIN); Kugluk (Bloody Falls) Territorial Park, flats above boardwalk W of Bloody Falls, 67°44′34.5″N, 115°22′27″W ± 100 m, 135 m, 16 July 2014, Saarela, Sokoloff & Bull 4058 (CAN); Kugluk (Bloody Falls) Territorial Park, along wet, muddy, and deeply pitted ATV trail ca. 1 km W of Bloody Falls, 67°44′33.2″N, 115°23′30″W ± 20 m, 73 m, 16 July 2014, Saarela, Sokoloff & Bull 4100 (CAN, MT, UBC); clay slopes and beach on E side of Coppermine River, just above start of Bloody Falls, 67°44′9.4″N, 115°22′41.2″W ± 15 m, 40 m, 19 July 2014, Saarela, Sokoloff & Bull 4215 (CAN); W of Kugluktuk on tundra flats above Coppermine River, S of 1 Coronation Drive and N of power plant, 67°49′28.97″N, 115°5′0.2″W ± 100 m, 8 m, 25 July 2014, Saarela, Sokoloff & Bull 4388 (ALA, CAN).

Puccinellia arctica (Hook.) Fernald & Weath., Figs. 35A and 35B—Arctic alkali grass | American Beringian–North American | Noteworthy Record

Figure 35 Puccinellia arctica and Puccinellia nuttalliana.

Puccinellia arctica: (A) habitat, Saarela et al. 4088. (B) Habit, Saarela et al. 4088. Puccinellia nuttalliana: (C) habit, Saarela et al. 4126. (D) Inflorescence, Saarela et al. 3797. Photographs by P. C. Sokoloff (A, B, C) and R. D. Bull (D).

Newly recorded for the study area. Our collections, all from the Bloody Falls area, close a distribution gap between the first mainland Nunavut collection from the Bathurst Inlet area (Cody, Scotter & Zoltai, 1984), reported as Puccinellia agrostidea Th. Sør, and Tuktut Nogait National Park and vicinity (Saarela et al., 2013a). One or more records from the Hood River (as Puccinellia agrostidea) in the Bathurst Inlet area have also been reported (Gould & Walker, 1997). Further west there are multiple collections from the Arctic coast of Northwest Territories and Yukon (Porsild & Cody, 1980; Davis & Consaul, 2007). In the Canadian Arctic Archipelago recorded from Banks, Ellesmere and Victoria islands (Aiken et al., 2007). The current concept of this species includes Puccinellia agrostidea and Puccinellia poacea Th. Sør. (Consaul & Gillespie, 2001; Davis & Consaul, 2007; Elven et al., 2011), recognised as distinct species in Porsild & Cody (1980). It was common on sparsely vegetated clay slopes and flats in the vicinity of Bloody Falls, growing with Achillea millefolium subsp. borealis, Chamerion latifolium, Deschampsia cespitosa, Festuca rubra subsp. arctica, Hedysarum boreale subsp. mackenziei, Hordeum jubatum subsp. intermedium and Puccinellia nuttalliana.

Specimens Examined: Canada. Nunavut: Kitikmeot Region: Kugluk (Bloody Falls) Territorial Park, S-facing clay slopes in gully on W side of Coppermine River, 67°44′12″N, 115°23′16.5″W ± 5 m, 41 m, 14 July 2014, Saarela, Sokoloff & Bull 4011 (CAN, MO, MT, O, UBC, US, WIN); Kugluk (Bloody Falls) Territorial Park, terrace above S-facing slopes above start of Bloody Falls, W side of Coppermine River, 67°44′27.2″N, 115°22′58″W ± 50 m, 68 m, 16 July 2014, Saarela, Sokoloff & Bull 4088 (ALA, ALTA, CAN, UBC, US); rocky beach along SE side of Coppermine River, above start of Bloody Falls, 67°44′16″N, 115°22′73.3″W ± 20 m, 17 m, 19 July 2014, Saarela, Sokoloff & Bull 4221a (CAN).

Puccinellia nuttalliana (Schult.) Hitchc., Figs. 35C and 35D—Nuttall’s alkali grass | Amphi-Pacific/Beringian–North American | Noteworthy Record

Newly recorded for the study area, where first collected in 2013 near the Kugluktuk Airport. We made collections at Fockler Creek, Kugluk (Bloody Falls) Territorial Park and Kugluktuk. These records fill in a distribution gap between Bathurst Inlet (Young’s Bay, 6 August 1950, Kelsall & McEwen 235, CAN-202973), Hood River (as Puccinellia borealis Swallen) (Gould & Walker, 1997) and eastern Great Bear Lake in the range maps (for Puccinellia borealis and Puccinellia deschampsioides Th. Sør.) of Porsild & Cody (1980). This is the most common Puccinellia species in the study area. It is highly variable with numerous forms that may look like different taxa in the field. It can be difficult to distinguish from Puccinellia vaginata, which also occurs in the study area, and a key to separate the two is given below. We found it growing in large dense populations on extensive clay slopes, where it was often the only plant species present. In one such habitat it was growing with Puccinellia arctica. The study area is near the northern limit of the species’ its range in northwestern Canada; to the north it is known only from Victoria Island (Aiken et al., 2007). There are few other collections from mainland Nunavut, with the exception of the extreme southeast (Davis & Consaul, 2007; e.g., Arviat, 11 July 2000, Boles RB00-62, CAN-600911; Rankin Inlet, Kudlulik Peninsula, 13 August 1974, Ohenoja 3, CAN-587999). In central mainland Nunavut recorded from Bathurst Inlet (Porsild & Cody, 1980); this collection is not mapped in Davis & Consaul (2007). Two species, Puccinellia deschampsioides Th. Sør. and Puccinellia interior Th. Sør, recognised in Porsild & Cody (1980) are treated as synonyms of Puccinellia nuttalliana by Davis & Consaul (2007) and Elven et al. (2011). A third species, Puccinellia borealis, is recognised by Porsild & Cody (1980) and Elven et al. (2011), but included in Puccinellia nuttalliana by Davis & Consaul (2007), whose treatment we follow. The geographic distribution given above combines that of Puccinellia nuttalliana and Puccinellia borealis from Elven et al. (2011).

Specimens Examined: Canada. Nunavut: Kitikmeot Region: Kugluktuk, airport, 21 July 2013, 67.81749°N, 115.13449°W, B. A. Bennett 13-0327a (CAN); E side of Fockler Creek, just above its confluence with Coppermine River, ca. 1.1 km SW of Sandstone Rapids, 67°26′30.6″N, 115°39′4.3″W ± 50 m, 135 m, 4 July 2014, Saarela, Sokoloff & Bull 3371a (CAN, MO, MT, O); confluence of Sleigh Creek and Coppermine River, 0.4 km N of Sandstone Rapids, 67°27′13.9″N, 115°38′7.7″W ± 25 m, 126 m, 6 July 2014, Saarela, Sokoloff & Bull 3481 (CAN, K, MO, MT, UBC, US, WIN); Kugluk (Bloody Falls) Territorial Park, rocky beach above Bloody Falls, W bank of Coppermine River, 67°44′18″N, 115°22′57.3″W ± 250 m, 34 m, 14 July 2014, Saarela, Sokoloff & Bull 3979 (ALA, ALTA, CAN, K, NY, UBC); Kugluk (Bloody Falls) Territorial Park, S-facing clay slopes in gully on W side of Coppermine River, 67°44′12″N, 115°23′16.5″W ± 5 m, 41 m, 14 July 2014, Saarela, Sokoloff & Bull 4010 (CAN, QFA, UBC, US, WIN); Kugluk (Bloody Falls) Territorial Park, terrace above S-facing slopes above start of Bloody Falls, W side of Coppermine River, 67°44′27.2″N, 115°22′58″W ± 50 m, 68 m, 16 July 2014, Saarela, Sokoloff & Bull 4093 (ALA, ALTA, CAN, K, NY, QFA, US, WIN); Kugluk (Bloody Falls) Territorial Park, rocky sandy beach just below Bloody Falls, W side of Coppermine River, vicinity of confluence with small creek, beach seasonally flooded, 67°44′54.5″N, 115°22′17.2″W ± 75 m, 9 m, 17 July 2014, Saarela, Sokoloff & Bull 4126 (CAN); Kugluk (Bloody Falls) Territorial Park, rocky sandy beach just below Bloody Falls, W side of Coppermine River, vicinity of confluence with small creek, beach seasonally flooded, 67°44′54.5″N, 115°22′17.2″W ± 75 m, 9 m, 17 July 2014, Saarela, Sokoloff & Bull 4127 (CAN, UBC); Kugluk (Bloody Falls) Territorial Park, rocky sandy beach just below Bloody Falls, W side of Coppermine River, vicinity of confluence with small creek, beach seasonally flooded, 67°44′54.5″N, 115°22′17.2″W ± 75 m, 9 m, 17 July 2014, Saarela, Sokoloff & Bull 4128 (CAN, MO, MT, O, QFA, WIN); Kugluk (Bloody Falls) Territorial Park, rocky sandy beach just below Bloody Falls, W side of Coppermine River, vicinity of confluence with small creek, beach seasonally flooded, 67°44′54.5″N, 115°22′17.2″W ± 75 m, 9 m, 17 July 2014, Saarela, Sokoloff & Bull 4130 (CAN, US); rocky beach along SE side of Coppermine River, above start of Bloody Falls, 67°44′16″N, 115°22′73.3″W ± 20 m, 17 m, 19 July 2014, Saarela, Sokoloff & Bull 4221b (ALA, ALTA, CAN); gravel roadside SW of Kugluktuk, S side of road to Heart Lake cemetery, just beyond sewage retention pond, 5.59 km SW of mouth of Coppermine River, 67°48′39″N, 115°12′38.7″W ± 25 m, 46 m, 23 July 2014, Saarela, Sokoloff & Bull 4323 (ALA, ALTA, CAN, O, UBC, US); Kugluktuk, ball diamond on S side of town, 67°49′23.2″N, 115°6′31.9″W ± 5 m, 29 m, 26 July 2014, Saarela, Sokoloff & Bull 4399 (ALA, ALTA, CAN, MO, MT, O, UBC).

Key to Puccinellia nuttalliana and Puccinellia vaginata (adapted from Davis & Consaul, 2007): 1. Culms usually erect; blades 1–4 mm wide, flat to involute; panicles 5–30 cm, usually exserted from sheath; inflorescence branches spikelet bearing from the base or on distal 2/3; pedicels scabrous, lacking tumid epidermal cells; lower glumes 0.5–1.5 mm; lemmas (2–)2.2–3(–3.5) mmPuccinellia nuttalliana

1′ Culms erect or decumbent; blades 1–2 mm wide when flat, 0.5–1.6 mm in diameter when involute; panicles (3–)6–12(–14) cm, usually barely exserted from sheath at maturity, spikelets usually confined to distal 2/3 of inflorescence branches; pedicels slightly scabrous, epidermal cells tumid; lower glumes 1.3–2.1 mm; lemmas 2.8–4 mmPuccinellia vaginata

Puccinellia phryganodes subsp. neoarctica (Á. Löve & D. Löve) Elven—Goosegrass | North American (N)

Previously reported for the study area (Porsild & Cody, 1980), but we were unable to locate a voucher specimen. We made only one collection of this halophytic species, from the tidal zone at Richardson Bay, where it was growing with Carex subspathacea, Potentilla anserina, Puccinellia vaginata and Stellaria humifusa. Our plants are the triploid race of Puccinellia phryganodes (Trin.) Scribn. & Merr. common along the Canadian Arctic coastline (Porsild & Cody, 1980; Davis & Consaul, 2007). Widespread throughout the Canadian Arctic Archipelago and recorded from several mainland sites (Porsild & Cody, 1980; Cody, Scotter & Zoltai, 1989; Korol, 1992; Aiken et al., 2007; Saarela et al., 2013a).

Specimens Examined: Canada. Nunavut: Kitikmeot Region: Richardson Bay, confluence of Richardson and Rae rivers at Coronation Gulf, ca. 20 km WNW of Kugluktuk, 67°54′11.2″N, 115°32′27.4″W ± 200 m, 0 m, 8 July 2014, Saarela, Sokoloff & Bull 3667a (ALA, ALTA, CAN, MO, O, UBC, US).

Puccinellia vaginata (Lange) Fernald & Welsh—Sheathed alkali grass | Amphi-Beringian–North American (N) | Noteworthy Record

Newly recorded for the study area. First collected in the study area around the Kugluktuk Airport in 2013 and we made three collections at Richardson Bay along the edge of the estuary and in wet trenches. Associated species at this site are listed under Puccinellia phryganodes, which was also common there. The only other record from mainland Nunavut appears to be from just west of the study area (Porsild & Cody, 1980; Davis & Consaul, 2007), though we were unable to locate a voucher. Elsewhere in the Canadian Arctic recorded from Baffin, Devon, Ellesmere, Southampton and Victoria islands and northern Quebec, with a large gap in the Central Arctic (Porsild & Cody, 1980; Aiken et al., 2007; Davis & Consaul, 2007; Saarela et al., 2013a). It is not clear if this gap is real or a sampling artefact.

Specimens Examined: Canada. Nunavut: Kitikmeot Region: Kugluktuk, airport, 21 July 2013, 67.81749°N, 115.13449°W, B. A. Bennett 13-0327b (CAN, det. B. A. Bennett, May 2014); Richardson Bay, confluence of Richardson and Rae rivers at Coronation Gulf, ca. 20 km WNW of Kugluktuk, 67°54′11.2″N, 115°32′27.4″W ± 200 m, 0 m, 8 July 2014, Saarela, Sokoloff & Bull 3671 (ALA, ALTA, CAN, MO, O); Richardson Bay, confluence of Richardson and Rae rivers at Coronation Gulf, ca. 20 km WNW of Kugluktuk, 67°54′11.2″N, 115°32′27.4″W ± 200 m, 0 m, 8 July 2014, Saarela, Sokoloff & Bull 3674 (CAN, MT, NY, QFA, WIN); Richardson Bay, confluence of Richardson and Rae rivers at Coronation Gulf, ca. 20 km WNW of Kugluktuk, 67°54′11.2″N, 115°32′27.4″W ± 200 m, 0 m, 8 July 2014, Saarela, Sokoloff & Bull 3667b (CAN, UBC, US).

Trisetum spicatum (L.) K. Richt., Fig. 36A—Narrow false-oat | Circumpolar-alpine

Figure 36 Trisetum spicatum and Stuckenia pectinata.

Trisetum spicatum: (A) habit, Saarela et al. 3376. Stuckenia pectinata: (B) habitat, Saarela et al. 4277. Photographs by J. M. Saarela (A) and P. C. Sokoloff (B).

Previously reported for the study area (Porsild & Cody, 1980), but we were unable to locate a voucher specimen. We made collections at Kugluktuk, Fockler Creek and Kugluk (Bloody Falls) Territorial Park. Widespread throughout the Canadian Arctic (Porsild & Cody, 1980; Cody, Scotter & Zoltai, 1989; Korol, 1992; Cody & Reading, 2005; Aiken et al., 2007; Saarela et al., 2013a).

Specimens Examined: Canada. Nunavut: Kitikmeot Region: Kugluktuk, airport, 21 July 2013, 67.81749°N, 115.13449°W, B. A. Bennett 13-0329 (BABY, det. B. A. Bennett, July 2013); Kugluktuk, rocky slopes of North Hill, 67°49′31.4″N, 115°6′54″W ± 100 m, 42 m, 29 June 2014, Saarela, Sokoloff & Bull 3078 (CAN, UBC); S of Fockler Creek, along small tributary that runs into Fockler Creek, ca. 2.3 km S of Sandstone Rapids, Coppermine River, 67°25′44.9″N, 115°38′25.9″W ± 100 m, 152 m, 3 July 2014, Saarela, Sokoloff & Bull 3273 (CAN); E side of Fockler Creek, just above its confluence with Coppermine River, ca. 1.1 km SW of Sandstone Rapids, 67°26′30.6″N, 115°39′4.3″W ± 50 m, 135 m, 4 July 2014, Saarela, Sokoloff & Bull 3376 (ALTA, CAN, O); confluence of Coppermine River and Melville Creek, just W of Coppermine Mountains, 67°15′52″N, 115°30′55.3″W ± 350 m, 178–190 m, 7 July 2014, Saarela, Sokoloff & Bull 3505 (ALA, CAN); S-facing sandstone cliffs above Coppermine River, ca. 7.8 km NNE of Sandstone Rapids, 67°31′15.1″N, 115°36′19.1″W ± 50 m, 106 m, 8 July 2014, Saarela, Sokoloff & Bull 3655 (CAN); Kugluk (Bloody Falls) Territorial Park, rocky cliffs and ledges directly above (W side) of Bloody Falls, just S of heavily used day-use/fishing area, 67°44′40.1″N, 115°22′4.9″W ± 20 m, 8 m, 12 July 2014, Saarela, Sokoloff & Bull 3807 (CAN, MO, MT); Kugluk (Bloody Falls) Territorial Park, W side of Coppermine River, between Sandy Hills and Bloody Falls, 67°45′13.2″N, 115°22′6.3″W ± 3 m, 21 m, 17 July 2014, Saarela, Sokoloff & Bull 4144 (CAN, US, WIN); W of Kugluktuk on tundra flats above Coppermine River, S of 1 Coronation Drive and N of power plant, 67°49′28.97″N, 115°5′0.2″W ± 100 m, 8 m, 25 July 2014, Saarela, Sokoloff & Bull 4365 (CAN, K, NY, QFA).

Potamogetonaceae [2/4]

Potamogeton gramineus L.—Grass-leaved pondweed | Circumboreal-polar | Noteworthy Record

Newly recorded for the study area, and a northeastern range extension from the Great Bear Lake area. The plants were locally common and submersed in about one ft. of water along the edge of the Coppermine River in Kugluktuk. Previously recorded on mainland Nunavut from the Thelon River (ca. 28 miles SW of Beverly Lake, ca. 64°36′N, 100°30′W, 23 July 1960, E. Kuyt 83, CAN-561543, det. R. R. Haynes 1995), as mapped in Porsild & Cody (1980) and Haynes & Hellquist (2000b). Elsewhere in the territory recorded from Akimiski Island (Blaney & Kotanen, 2001). In their range map, Haynes & Hellquist (2000b) also included a small sliver of southern Nunavut adjacent to the Manitoba and Saskatchewan border, but it is unclear if its occurrence in this area is documented by specimens. The taxon sometimes has floating leaves, but none was present in the population from which our collection was gathered. It also reaches the Arctic in adjacent Northwest Territories (e.g., Anderson River Delta, 69°42′N, 129°W, T. W. Barry 192, CAN-287856, det. R. R. Haynes 1995).

Specimens Examined: Canada. Nunavut: Kitikmeot Region: W of Kugluktuk on tundra flats above Coppermine River, S of 1 Coronation Drive and N of community power plant, 67°49′28.97″N, 115°5′0.2″W ± 100 m, 8 m, 22 July 2014, Saarela, Sokoloff & Bull 4276 (ALA, ALTA, CAN, UBC).

Stuckenia filiformis (Pers.) Börner—Slender-leaved pondweed | Circumboreal–polar | Noteworthy Record

Newly recorded for the study area. We found it in a small shallow pond on an island in the mouth of the Coppermine River, and at two sites in Kugluk (Bloody Falls) Territorial Park. At one site the plants were uncommon and rooted in wet moss at the edge of a pond, previously underwater, while at the other site plants were locally common in wet sand at the edge of the Coppermine River, in an area that had probably recently dried up. It was recently recorded from a site just outside the study area (Reading 50, DAO; Cody & Reading, 2005). These collections close a distribution gap between Bathurst Inlet and eastern Great Bear Lake (Porsild & Cody, 1980). Elsewhere in the Canadian Arctic recorded from Baffin and Southampton islands, a few sites along western Hudson Bay and northern Quebec (Korol, 1992; Aiken et al., 2007; Garneau, 2013a).

The taxonomy of Stuckenia Börner is complex and there are several conflicting treatments (Tolmachev & Packer, 1995; Haynes & Hellquist, 2000b; Kaplan, 2008; see discussion in Elven et al., 2011). We follow the revisionary treatment of Kaplan (2008), who does not recognise subspecies in this variable species and includes an excellent key. Our collections key to Stuckenia filiformis subsp. alpina (Blytt) R. R. Haynes, Les & M. Král in Haynes & Hellquist (2000b). In Elven et al. (2011) and Aiken et al. (2007) they correspond to Stuckenia filiformis subsp. borealis (Raf.) Tzvelev & Elven. Elven et al. (2011) recognised subsp. borealis as an Amphi-Beringian and North American taxon, and treated European and Asian plants as Stuckenia filiformis subsp. filiformis (including Stuckenia filiformis subsp. alpina as a synonym), a taxon Haynes & Hellquist (2000b) recognised as occurring in northwestern North America and Greenland.

The circumscription of Stuckenia filiformis subsp. occidentalis (J. W. Robbins) R. R. Haynes, Les & M. Král—a name based on specimens from Nevada—is problematic. Haynes & Hellquist (2000b) recognised it as widespread across North America, and it was recently recognised in a treatment of the flora of northern Quebec (Garneau, 2013a). Elven et al. (2011) note that plants treated under this name resemble Stuckenia pectinata and Stuckenia vaginata more than Stuckenia filiformis in several characters (Elven et al., 2011). Kaplan (2008) did not formally treat this name in his revision of the genus in Asia, but noted that many individuals treated under this name have open sheaths (vs. closed)—a character he emphasised as distinguishing Stuckenia filiformis (closed sheaths) from Stuckenia pectinata and Stuckenia vaginata (open sheaths)—and they therefore could not be considered to be Stuckenia filiformis. Haynes & Hellquist (2000b) did not include the nature of the leaf sheath as a character in their keys or descriptions. Kaplan (2008) also noted the possibility they may represent hybrids between Stuckenia pectinata and Stuckenia vaginata.

Specimens Examined: Canada. Nunavut: Kitikmeot Region: unnamed island just E (ca. 3.3 km) of Kugluktuk at mouth of Coppermine River, 67°49′29.2″N, 115°1′3.2″W ± 50 m, 1 m, 8 July 2014, Saarela, Sokoloff & Bull 3723 (CAN); Kugluk (Bloody Falls) Territorial Park, rocky valley immediately SW of Bloody Falls, along rough marked section of Portage Trail, 67°44′34″N, 115°22′16″W ± 50 m, 20 m, 13 July 2014, Saarela, Sokoloff & Bull 3881 (CAN); Kugluk (Bloody Falls) Territorial Park, rocky valley immediately SW of Bloody Falls, along rough marked section of Portage Trail, 67°44′34″N, 115°22′16″W ± 50 m, 20 m, 13 July 2014, Saarela, Sokoloff & Bull 3882 (CAN); Kugluk (Bloody Falls) Territorial Park, rocky sandy beach just below Bloody Falls, W side of Coppermine River, vicinity of confluence with small creek, beach seasonally flooded, 67°44′54.5″N, 115°22′17.2″W ± 75 m, 9 m, 17 July 2014, Saarela, Sokoloff & Bull 4136 (CAN).

Stuckenia pectinata (L.) Börner, Fig. 36B—Sago pondweed | Circumboreal | Noteworthy Record

Newly recorded for the study area and Nunavut, and a northeastern range extension. We made two collections of this boreal species above Bloody Falls and one below; two of these sites are in Kugluk (Bloody Falls) Territorial Park. At each site it was locally common and submersed in 0.5–1 feet of water along the edge of the Coppermine River. We found it growing in a similar habitat along the Coppermine River at Kugluktuk. The closest collections are from the Mackenzie River Delta (Porsild, 1943; Porsild & Cody, 1980; specimens at CAN); these were not mapped in Haynes & Hellquist (2000b), who may have assigned these plants to Stuckenia filiformis subsp. occidentalis (Elven et al., 2011), which they mapped for the Mackenzie Delta. Not recorded from other sites in the Canadian Arctic.

Specimens Examined: Canada. Nunavut: Kitikmeot Region: Kugluk (Bloody Falls) Territorial Park, rocky beach above Bloody Falls, W bank of Coppermine River, 67°44′18″N, 115°22′57.3″W ± 250 m, 34 m, 14 July 2014, Saarela, Sokoloff & Bull 3980 (ALA, ALTA, CAN); Kugluk (Bloody Falls) Territorial Park, rocky sandy beach just below Bloody Falls, W side of Coppermine River, vicinity of confluence with small creek, beach seasonally flooded, 67°44′54.5″N, 115°22′17.2″W ± 75 m, 9 m, 17 July 2014, Saarela, Sokoloff & Bull 4135 (CAN); rocky beach along SE side of Coppermine River, above start of Bloody Falls, 67°44′16″N, 115°22′73.3″W ± 20 m, 17 m, 19 July 2014, Saarela, Sokoloff & Bull 4224 (CAN, UBC); W of Kugluktuk on tundra flats above Coppermine River, S of 1 Coronation Drive and N of community power plant, 67°49′28.97″N, 115°5′0.2″W ± 100 m, 8 m, 22 July 2014, Saarela, Sokoloff & Bull 4277 (CAN, MO, O).

Stuckenia vaginata (Turcz.) Holub.—Big-sheath pondweed | European (N) & Asian (C-NE) & North American | Noteworthy Record

Newly recorded for the study area and western mainland Nunavut. We found this primarily boreal species at three sites: near the Kendall River, on an island in the mouth of the Coppermine River and in Kugluk (Bloody Falls) Territorial Park. Known from a few collections from the mainland Canadian Arctic in Northwest Territories and southeastern Nunavut (Porsild, 1950b; Porsild & Cody, 1980; Cody, Reading & Line, 2003; Saarela et al., 2013a), and recently reported from a site on Victoria Island, Nunavut, the first record for the Canadian Arctic Archipelago (Gillespie et al., 2015).

Specimens Examined: Canada. Nunavut: Kitikmeot Region: confluence of Coppermine and Kendall rivers (NW side of Coppermine River, S side of Kendall River), small ponds on flats adjacent to Coppermine River, 67°6′44.7″N, 116°8′6.1″W ± 100 m, 213 m, 7 July 2014, Saarela, Sokoloff & Bull 3589 (CAN); unnamed island just E (ca. 3.3 km) of Kugluktuk at mouth of Coppermine River, 67°49′29.2″N, 115°1′3.2″W ± 50 m, 1 m, 8 July 2014, Saarela, Sokoloff & Bull 3722 (CAN); Kugluk (Bloody Falls) Territorial Park, rocky valley immediately SW of Bloody Falls, along rough marked section of Portage Trail, 67°44′34″N, 115°22′16″W ± 50 m, 20 m, 13 July 2014, Saarela, Sokoloff & Bull 3880 (CAN).

Tofieldiaceae [1/2]

Tofieldia coccinea Richardson, Fig. 37—Pink-flowered asphodel, Northern tofieldia | Asian (N/C)–amphi-Beringian–North American (N)

Figure 37 Tofieldia coccinea.

(A) Habit, Saarela et al. 3770. (B) Inflorescence, Saarela et al. 3770. (C) Habitat, Saarela et al. 3770. Photographs by R. D. Bull.

Previously recorded from Kugluktuk (Cody, 1954b; Porsild & Cody, 1980). We made collections at Fockler Creek, Coppermine Mountains, Kugluk (Bloody Falls) Territorial Park and Kugluktuk. A key to separate Tofieldia coccinea and Tofieldia pusilla is given in Saarela et al. (2013a). Elsewhere in the Canadian Arctic recorded from Banks, Baffin, Devon and Victoria islands, and some mainland sites (Porsild & Cody, 1980; Cody, Scotter & Zoltai, 1989; Korol, 1992; Cody & Reading, 2005; Aiken et al., 2007; Gauthier, 2013; Saarela et al., 2013a).

Specimens Examined: Canada. Nunavut: Kitikmeot Region: Coppermine [Kugluktuk], 67°49′36″N, 115°5′36″W, 1 August 1951, W. I. Findlay 232 (DAO-176341 01-01000677356); top of ridge N of Fockler Creek, ca. 3.6 km SE of Sandstone Rapids, Coppermine River, 67°25′20″N, 115°35′40.1″W ± 5 m, 235 m, 5 July 2014, Saarela, Sokoloff & Bull 3418 (CAN, MO, O); forest and slopes at confluence of Big Creek and Coppermine River, N side of Coppermine River, S side of Coppermine Mountains, 67°14′29.3″N, 116°2′44.5″W ± 250 m, 180–199 m, 7 July 2014, Saarela, Sokoloff & Bull 3546 (CAN, MT, US); flats atop and upper slopes of Coppermine Mountains, N/W side of Coppermine River, 67°36′58.7″N, 115°29′18.3″W ± 99 m, 50 m, 8 July 2014, Saarela, Sokoloff & Bull 3783 (CAN); flats atop and upper slopes of Coppermine Mountains, N/W side of Coppermine River, 67°14′49.9″N, 115°38′43.7″W ± 200 m, 467 m, 9 July 2014, Saarela, Sokoloff & Bull 3770 (CAN, UBC); Kugluk (Bloody Falls) Territorial Park, upper ledges of rocky (gabbro) S-facing cliffs above the start of Bloody Falls (W bank of River), just E of Portage Trail, 67°44′21.7″N, 115°22′42.2″W ± 25 m, 46 m, 14 July 2014, Saarela, Sokoloff & Bull 3945 (CAN); Kugluk (Bloody Falls) Territorial Park, flats on top of mountain on W side of Coppermine River, just S of the start of Bloody Falls Rapids, 67°44′2.8″N, 115°23′39.3″W ± 250 m, 110 m, 14 July 2014, Saarela, Sokoloff & Bull 3994 (ALA, CAN); W of Kugluktuk on tundra flats above Coppermine River, S of 1 Coronation Drive and N of power plant, 67°49′28.97″N, 115°5′0.2″W ± 100 m, 8 m, 25 July 2014, Saarela, Sokoloff & Bull 4378 (ALTA, CAN).

Tofieldia pusilla (Michx.) Pers.—Bog asphodel, small tofieldia | Circumpolar–alpine

Previously recorded from Kugluktuk (Cody, 1954b; Porsild & Cody, 1980). We made collections at Fockler Creek, Big Creek, Kugluk (Bloody Falls) Territorial Park and Kugluktuk. Elsewhere in the Canadian Arctic recorded from Baffin, Southampton and Victoria islands, and several mainland sites (Porsild & Cody, 1980; Cody, Scotter & Zoltai, 1989; Korol, 1992; Cody & Reading, 2005; Aiken et al., 2007; Gauthier, 2013; Saarela et al., 2013a; Bennett, 2015).

Specimens Examined: Canada. Nunavut: Kitikmeot Region: Coppermine [Kugluktuk], 67°49′36″N, 115°5′36″W, 8 July 1951, W. I. Findlay 103 (DAO-176422 01-01000677807); Kugluktuk, overlooking Coppermine River [67°49′36″N, 115°5′36″W], 16 July 2000, L. K. Benjamin s.n. (ACAD-ECS015890); Kugluktuk, rocky slopes of North Hill, 67°49′31.4″N, 115°6′54″W ± 100 m, 42 m, 29 June 2014, Saarela, Sokoloff & Bull 3068 (CAN); old riverbed of Fockler Creek, ca. 2.3 km SSE of Sandstone Rapids, Coppermine River, 67°25′45.7″N, 115°37′21.8″W ± 25 m, 166 m, 1 July 2014, Saarela, Sokoloff & Bull 3153 (CAN, UBC); S of Fockler Creek, along small tributary that runs into Fockler Creek, ca. 2.3 km S of Sandstone Rapids, Coppermine River, 67°25′44.9″N, 115°38′25.9″W ± 100 m, 152 m, 3 July 2014, Saarela, Sokoloff & Bull 3241 (ALA, CAN); forest and slopes at confluence of Big Creek and Coppermine River, N side of Coppermine River, S side of Coppermine Mountains, 67°14′29.3″N, 116°2′44.5″W ± 250 m, 180–199 m, 7 July 2014, Saarela, Sokoloff & Bull 3547 (ALTA, CAN); Kugluk (Bloody Falls) Territorial Park, rocky valley immediately SW of Bloody Falls, along rough marked section of Portage Trail, 67°44′34″N, 115°22′16″W ± 50 m, 20 m, 13 July 2014, Saarela, Sokoloff & Bull 3868 (CAN, O); W of Kugluktuk on tundra flats above Coppermine River, S of 1 Coronation Drive and N of power plant, 67°49′28.97″N, 115°5′0.2″W ± 100 m, 8 m, 25 July 2014, Saarela, Sokoloff & Bull 4382 (CAN).

Typhaceae [1/1]

Sparganium hyperboreum Laest. ex Beurl., Fig. 38—Northern burreed | Circumboreal-polar | Noteworthy Record

Figure 38 Sparganium hyperboreum.

(A) Inflorescence, Saarela et al. 3617. (B) Habitat, Saarela et al. 3617. (C) Habit, Saarela et al. 3617. Photographs by P. C. Sokoloff (A, B) and J. M. Saarela (C).

First record of the species for the study area, where we encountered a dense stand in a small pond, growing with Carex saxatilis, at a site south of Escape Rapids. The species is at the northern edge of its range in the study area. It was previously reported from an Arctic site outside the study area, ca. 30 km east of the Coppermine River (Reading 52, DAO; Cody & Reading, 2005). The nearest known sites to these collections are in the Great Bear Lake and Bathurst Inlet areas (Porsild, 1943; Porsild & Cody, 1980). Elsewhere in Nunavut recorded from Yathkyed Lake, the Tha-anne River and the Nueltin Lake area north of the Manitoba border (Porsild, 1943, 1950b; Porsild & Cody, 1980). It also reaches the Canadian Arctic in northern Quebec and Labrador (Kaul, 1997; Garneau, 2013b).

Specimens Examined: Canada. Nunavut: Kitikmeot Region: S-facing slopes on W side of Coppermine River, about halfway between Escape Rapids and Muskox Rapids, 67°31′18.2″N, 115°36′20.1″W ± 150 m, 115 m, 8 July 2014, Saarela, Sokoloff & Bull 3617 (ALA, ALTA, CAN, O, UBC).

Eudicots

Asteraceae [15/23]

Achillea millefolium subsp. borealis (Bong.) Breitung, Fig. S15—Common yarrow | Amphi-Beringian–North American (N)

Previously recorded from Bloody Falls and Kugluktuk (Cody, 1954b, as Achillea millefolium subsp. atrotegula B. Boivin; Porsild & Cody, 1980). We made collections in Kugluk (Bloody Falls) Territorial Park and Kugluktuk. Achillea millefolium is a widespread, morphologically variable species that has been variously treated taxonomically. Trock (2006) did not recognise infraspecific taxa in North America. Elven et al. (2011) tentatively treated northern North American plants as Achillea millefolium subsp. borealis, the classification we follow. Porsild & Cody (1980) recognised native northern plants as Achillea nigrescens (E. Mey.) Rydb. and plants introduced from the Old World as Achillea millefolium L. (=Achillea millefolium subsp. millefolium in Elven et al., 2011). Achillea nigrescens is treated as a synonym of Achillea millefolium s.l. in Trock (2006) and of Achillea millefolium subsp. borealis in Elven et al. (2011). Elsewhere in the Canadian Arctic recorded from Banks Island and southern Baffin Island, and on the mainland from the Bathurst Inlet area and to the west (Aiken et al., 2007; Bennett, 2015). The species is at the northern edge of its range in the study area.

Specimens Examined: Canada. Nunavut: Kitikmeot Region: Bloody Falls, 67°44′N, 115°23′W, 18 July 1951, W. I. Findlay 149 (DAO 01-01000616612); Coppermine [Kugluktuk], 67°49′36″N, 115°5′36″W, 21 July 1951, W. I. Findlay 159 (ALTA-VP-16422, DAO 01-01000616613, UBC-V40758); Coppermine [Kugluktuk], 67°49′36″N, 115°5′36″W, 2 August 1995, T. Dolman 98 (LEA), Kugluktuk, 67.82125°N, 115.085217°W, 6 July 2006, J. Davis 618 (CAN-597644); Kugluk (Bloody Falls) Territorial Park, rocky beach above Bloody Falls, W bank of Coppermine River, 67°44′18″N, 115°22′57.3″W ± 250 m, 34 m, 14 July 2014, Saarela, Sokoloff & Bull 3959 (ALA, ALTA, CAN); Kugluk (Bloody Falls) Territorial Park, terrace above S-facing slopes above start of Bloody Falls, W side of Coppermine River, 67°44′27.2″N, 115°22′58″W ± 50 m, 68 m, 16 July 2014, Saarela, Sokoloff & Bull 4090 (CAN, US); SSW-facing slopes above start of Bloody Falls, SE side of Coppermine River, 67°44′12.5″N, 115°22′31″W ± 50 m, 50–60 m, 19 July 2014, Saarela, Sokoloff & Bull 4208 (CAN, UBC); W of Kugluktuk on tundra flats above Coppermine River, S of 1 Coronation Drive and N of community power plant, 67°49′28.97″N, 115°5′0.2″W ± 100 m, 8 m, 22 July 2014, Saarela, Sokoloff & Bull 4249 (CAN, MT).

Antennaria friesiana (Trautv.) E. Ekman—Antennaria alpina (Trautv.) E. Ekman complex

Antennaria is an extremely taxonomically difficult genus due to extensive polyploidy, hybridisation and asexual reproduction (apomixis). Although considerable systematic investigation has been conducted on species groups that include Arctic plants (Chmielewski & Chinnappa, 1991; Bayer, 1993; Bayer & Stebbins, 1993; Chmielewski, 1994, 1997, 1998), the problems are confounded by competing taxonomic concepts. For example, Antennaria media subsp. compacta (Malte) Chmiel. (syn. Antennaria compacta Malte), treated as a distinct taxon by Chmielewski (1997) and recognised in recent treatments for the Canadian Arctic (Bayer, 2006; Aiken et al., 2007; Elven et al., 2011), is included in Antennaria alpina s.l. by Bayer (2006). Plants included here all have two or more heads (capitula), distinguishing them from Antennaria monocephala DC. (Bayer, 2006), and are likely referable to either Antennaria friesiana s.l., Antennaria alpina s.l. or Antennaria media subsp. compacta, depending on whose taxonomy is followed. All characters in the key in Bayer (2006) distinguishing Antennaria friesiana and Antennaria alpina overlap. Unfortunately, in none of his papers does Chmielewski provide a treatment for all taxa he recognises, and his papers treating different groups do not all indicate clearly how the groups differ from one another. Distinguishing the above-named taxa is exceptionally difficult for a non-expert, and here the plants are treated as a broadly circumscribed species complex pending further study. A collection from the mouth of the Napaaktoktok River (Findlay 184A, DAO) just east of the study area was published as Antennaria subcanescens Ostenf. ex Malte (Cody, 1954b) and re-determined as Antennaria compacta by A. E. Porsild in 1967.

Specimens Examined: Canada. Nunavut: Kitikmeot Region: Kugluktuk, rocky slopes of North Hill, 67°49′29.6″N, 115°6′31″W ± 50 m, 50 m, 29 June 2014, Saarela, Sokoloff & Bull 3059 (CAN, US); old riverbed of Fockler Creek, ca. 2.3 km SSE of Sandstone Rapids, Coppermine River, 67°25′48″N, 115°37′33″W ± 25 m, 153 m, 1 July 2014, Saarela, Sokoloff & Bull 3150 (CAN); third ridge N of Fockler Creek overlooking small lake, ca. 1.7 km SSE of Sandstone Rapids, Coppermine River, 67°26′7.3″N, 115°37′28″W ± 3 m, 171 m, 2 July 2014, Saarela, Sokoloff & Bull 3221 (ALTA, CAN); N side of Fockler Creek, ca. 1.9 km S of Sandstone Rapids, Coppermine River, 67°25′57.89″N, 115°38′3.9″W ± 10 m, 162 m, 4 July 2014, Saarela, Sokoloff & Bull 3332 (CAN); W shore of Tundra Lake, ca. 4.3 km SE of Sandstone Rapids, Coppermine River, 67°25′43.7″N, 115°33′6.8″W ± 4 m, 262 m, 5 July 2014, Saarela, Sokoloff & Bull 3443 (CAN, O); W shore of Tundra Lake, ca. 4.3 km SE of Sandstone Rapids, Coppermine River, 67°25′43.7″N, 115°33′6.8″W ± 4 m, 262 m, 5 July 2014, Saarela, Sokoloff & Bull 3445 (CAN, MO); S-facing slopes on W side of Coppermine River, about halfway between Escape Rapids and Muskox Rapids, 67°31′18.2″N, 115°36′20.1″W ± 150 m, 115 m, 8 July 2014, Saarela, Sokoloff & Bull 3628 (CAN); flats atop and upper slopes of Coppermine Mountains, NW side of Coppermine River, 67°14′49.9″N, 115°38′43.7″W ± 200 m, 467 m, 9 July 2014, Saarela, Sokoloff & Bull 3765 (CAN); flats atop and upper slopes of Coppermine Mountains, NW side of Coppermine River, 67°14′49.9″N, 115°38′43.7″W ± 200 m, 467 m, 9 July 2014, Saarela, Sokoloff & Bull 3779 (CAN); Kugluk (Bloody Falls) Territorial Park, rocky cliffs and ledges directly above (W side) of Bloody Falls, just S of heavily used day-use/fishing area, 67°44′40.1″N, 115°22′4.9″W ± 20 m, 8 m, 12 July 2014, Saarela, Sokoloff & Bull 3813 (CAN, MT); W of Kugluktuk on tundra flats above Coppermine River, S of 1 Coronation Drive and N of community power plant, 67°49′28.97″N, 115°5′0.2″W ± 100 m, 8 m, 22 July 2014, Saarela, Sokoloff & Bull 4240 (CAN, UBC).

Antennaria monocephala subsp. angustata (Greene) Hultén—Pygmy pussy-toes | Amphi-Beringian–North American (N)

Previously recorded from Kugluktuk (Cody, 1954b; Porsild & Cody, 1980). We made collections at Fockler Creek and in Kugluk (Bloody Falls) Territorial Park. No collections were mapped in the study area in Chmielewski & Chinnappa (1991), but collections were mapped for the taxon (as Antennaria angustata Greene, a synonym) in Porsild & Cody (1980). Elsewhere in the Canadian Arctic recorded from Banks, Devon, Coats, Melville and Nottingham Islands, and mainland sites (Porsild & Cody, 1980; Aiken et al., 2007; Saarela et al., 2013a).

Specimens Examined: Canada. Nunavut: Kitikmeot Region: Coppermine [Kugluktuk], 67°49′36″N, 115°5′36″W, 1 August 1951, W. I. Findlay 230 (DAO-177181 01-01000616991); Coppermine [Kugluktuk], 67°49′36″N, 115°5′36″W, 26 July 1951, W. I. Findlay 184B (DAO-177183 01-01000616990); W shore of Tundra Lake, ca. 4.3 km SE of Sandstone Rapids, Coppermine River, 67°25′43.7″N, 115°33′6.8″W ± 4 m, 262 m, 5 July 2014, Saarela, Sokoloff & Bull 3444 (CAN, US); flats atop and upper slopes of Coppermine Mountains, N/W side of Coppermine River, 67°14′43.7″N, 115°38′51.2″W ± 150 m, 422 m, 9 July 2014, Saarela, Sokoloff & Bull 3751 (CAN); Kugluk (Bloody Falls) Territorial Park, small rocky meadow along small stream that runs into Coppermine River just below Bloody Falls, about 1 km W of Bloody Falls, 67°44′40.1″N, 115°23′37.5″W ± 15 m, 48 m, 15 July 2014, Saarela, Sokoloff & Bull 4026 (CAN); Kugluk (Bloody Falls) Territorial Park, rocky valley immediately SW of Bloody Falls, along rough marked section of Portage Trail, 67°44′34″N, 115°22′16″W ± 50 m, 20 m, 18 July 2014, Saarela, Sokoloff & Bull 4160 (CAN); Kugluk (Bloody Falls) Territorial Park, rocky valley immediately SW of Bloody Falls, along rough marked section of Portage Trail, 67°44′34″N, 115°22′16″W ± 50 m, 20 m, 18 July 2014, Saarela, Sokoloff & Bull 4161 (CAN).

Arnica angustifolia Vahl subsp. angustifolia, Fig. S16—Alpine arnica | North American (N)–amphi-Atlantic (W)

Previously recorded from Kugluktuk (Cody, 1954b, as Arnica alpina var. vahliana B. Boivin; Porsild & Cody, 1980). We made collections at Fockler Creek, Kugluk (Bloody Falls) Territorial Park and Kugluktuk. Taxonomy follows Elven et al. (2011) and Wolf (2006), although the circumscription of the taxon in Wolf (2006) is broader, including Arnica angustifolia subsp. attenuata (Greene) G. W. Douglas & Ruyle-Douglas, which is recognised as a distinct taxon in Elven et al. (2011) and Porsild & Cody (1980, as Arnica alpina subsp. attenuata (Greene) Maguire). This taxon has also been recognised as Arnica alpina subsp. angustifolia (Vahl) Maguire (Porsild & Cody, 1980). In the Canadian Arctic recorded from Baffin, Banks, Ellesmere, Emerald, Melville, Somerset and Victoria islands, and mainland sites (Porsild & Cody, 1980; Cody & Reading, 2005; Aiken et al., 2007; Saarela et al., 2013a).

Specimens Examined: Canada. Nunavut: Kitikmeot Region: Coppermine [Kugluktuk], 67°49′36″N, 115°5′36″W, 27 July 1951, W. I. Findlay 40 (DAO-161697 01-01000616645); Kugluktuk, overlooking Coppermine River [67°49′36″N, 115°5′36″W], 16 July 2000, L. K. Benjamin s.n. (ACAD-ECS015860); Kugluktuk, rocky slopes of North Hill, 67°49′31.4″N, 115°6′54″W ± 100 m, 42 m, 29 June 2014, Saarela, Sokoloff & Bull 3069 (CAN); flats on W side of Fockler Creek, above spruce forest in creek valley, ca. 2.2 km S of Sandstone Rapids, Coppermine River, 67°25′49″N, 115°37′55″W ± 50 m, 152 m, 1 July 2014, Saarela, Sokoloff & Bull 3122 (CAN, MT); Kugluk (Bloody Falls) Territorial Park, rocky cliffs and ledges directly above (W side) of Bloody Falls, just S of heavily used day-use/fishing area, 67°44′40.1″N, 115°22′4.9″W ± 20 m, 8 m, 12 July 2014, Saarela, Sokoloff & Bull 3820 (CAN, US).

Arnica frigida C. A. Mey. ex Iljin, Fig. S17—Snow arnica | Amphi-Beringian

Previously recorded from Kugluktuk (Cody, 1954b; Porsild & Cody, 1980). We made collections at Big Creek, Kugluk (Bloody Falls) Territorial Park and Kugluktuk. Taxonomy follows Elven et al. (2011). It has also been recognised as Arnica griscomii subsp. frigida (C. A. Mey. ex Iljin) S. J. Wolf (Wolf, 2006) and Arnica louiseana subsp. frigida (C. A. Mey. ex Iljin) Maguire (Porsild & Cody, 1980). It extends to just west of Bathurst Inlet (Porsild & Cody, 1980). Wolf (2006) did not include Nunavut in its range even though collections were mapped in Porsild & Cody (1980) at sites within the territory. It is not recorded for the Canadian Arctic Archipelago and is at the edge of its range in the study area.

Specimens Examined: Canada. Nunavut: Kitikmeot Region: Coppermine [Kugluktuk], Coronation Gulf, at mouth of Coppermine River [67.822146°N, 115.078387°W ± 0.5 km], 4 August 1948, H. T. Shacklette 3298 (CAN-200101); Coppermine [Kugluktuk], 67°49′36″N, 115°5′36″W, W. I. Findlay 129 (DAO); Coppermine [Kugluktuk], south facing cliff overlooking Coppermine River [67.816375°N, 115.1002722°W ± 350 m], 11 July 1958, R. D. Wood s.n. (CAN-265449); forest and slopes at confluence of Big Creek and Coppermine River, N side of Coppermine River, S side of Coppermine Mountains, 67°14′29.3″N, 116°2′44.5″W ± 250 m, 180–199 m, 7 July 2014, Saarela, Sokoloff & Bull 3543 (CAN); Kugluk (Bloody Falls) Territorial Park, gentle stream in shallow valley running into Coppermine River just W of Bloody Falls, 67°44′36.6″N, 115°22′59.3″W ± 20 m, 41 m, 15 July 2014, Saarela, Sokoloff & Bull 4013 (CAN, US); W of Kugluktuk on tundra flats above Coppermine River, S of 1 Coronation Drive and N of power plant, 67°49′28.97″N, 115°5′0.2″W ± 100 m, 8 m, 25 July 2014, Saarela, Sokoloff & Bull 4371 (CAN, MT).

Artemisia borealis Pallas subsp. borealis—Boreal wormwood | European (NE)–Asian (N/C)–amphi-Beringian–Cordilleran–North American (N) | Noteworthy Record

Newly recorded for the study area. We made collections at Fockler and Melville creeks, and it was recently recorded from the Big Bend area of the Coppermine River (Reading 30, DAO; Cody, Reading & Line, 2003) and two sites west of the Coppermine River (67°33′28″N, 116°14′59″W, Reading 58, DAO; 67°38′N, 116°19′W, Reading 34, DAO). All of these collections close a distribution gap between Bathurst Inlet, Hood River, eastern Great Slave Lake and Tuktut Nogait National Park and vicinity (Porsild & Cody, 1980; Gould & Walker, 1997; Saarela et al., 2013a). Elsewhere in the Canadian Arctic recorded from southern Baffin Island, Banks and Victoria islands, and a few other mainland sites (Porsild & Cody, 1980; Aiken et al., 2007; Saarela et al., 2013a). Taxonomy follows Shultz (2006) and Elven et al. (2011) who recognise two subspecies; by contrast, Porsild & Cody (1980) recognised these taxa at species level. Subspecies borealis is a widespread circumboreal taxon that reaches the Low Arctic in Canada, while subsp. richardsoniana has an Amphi-Beringian distribution. The latter subspecies was recorded for the study area, as Artemisia richardsoniana Besser, in Porsild & Cody (1980). We were unable to locate a voucher specimen to confirm that record and did not collect the taxon in 2014. A key to distinguish subsp. borealis and subsp. richardsoniana is given in Saarela et al. (2013a). One of the collections treated here approaches subsp. richardsoniana, in having quite villous blades, but the hairs are not as dense as is typical for subsp. richardsoniana.

Specimens Examined: Canada. Nunavut: Kitikmeot Region: top of ridge N of Fockler Creek, ca. 3.6 km SE of Sandstone Rapids, Coppermine River, 67°25′20″N, 115°35′40.1″W ± 5 m, 235 m, 5 July 2014, Saarela, Sokoloff & Bull 3419 (CAN, US); confluence of Coppermine River and Melville Creek, just W of Coppermine Mountains, 67°15′52″N, 115°30′55.3″W ± 350 m, 178–190 m, 7 July 2014, Saarela, Sokoloff & Bull 3501 (CAN).

Artemisia hyperborea Rydb., Fig. S18—Northern wormwood | American Beringian

A previous collection from near Kugluktuk is the only record of this species for Nunavut mapped in Porsild & Cody (1980), but we were unable to locate a voucher specimen. It was recently recorded from a site ca. 20 km west of the study area (Cody & Reading, 2005). We made collections near Muskox Rapids and in Kugluk (Bloody Falls) Territorial Park. The species is at the eastern limit of its mainland Arctic range in the study area and is also recorded from Arctic sites on mainland Northwest Territories (Porsild & Cody, 1980; Saarela et al., 2013a). In the western Canadian Arctic Archipelago recorded from Victoria Island (Northwest Territories) and Banks Island (Aiken et al., 2007; L. J. Gillespie & J. M. Saarela, 2016, unpublished data). It was treated as a synonym of Artemisia furcate M. Bieberstein in Shultz (2006), but considered a distinct species in Elven et al. (2011).

Specimens Examined: Canada. Nunavut: Kitikmeot Region: esker on E side of Coppermine River, 0.6 km SSE of Muskox Rapids, 67°22′40″N, 115°42′38.5″W ± 50 m, 172 m, 7 July 2014, Saarela, Sokoloff & Bull 3611 (CAN); Kugluk (Bloody Falls) Territorial Park, sandy NE-facing slope above small creek in deep gully, about 0.5 km W of Bloody Falls, 67°44′36.6″N, 115°22′59.3″W ± 41 m, 41 m, 15 July 2014, Saarela, Sokoloff & Bull 4020 (CAN, UBC, US).

Artemisia tilesii Ledeb.—Tilesius’s wormwood | European (NE)–Asian (N)–amphi-Beringian–North American (N)

Previously recorded from Bloody Falls (Cody, 1954b; Porsild & Cody, 1980). It was collected in Kugluktuk in 1999 and we made collections at Fockler Creek, Melville Creek and in Kugluk (Bloody Falls) Territorial Park. This is a common species in disturbed areas along the Coppermine River and is distinguished from congeneric species in the study area by its distinctly rhizomatous habit. It reaches its known eastern limit in central mainland Nunavut, and in the Canadian Arctic Archipelago is recorded only from Banks and Victoria islands (Porsild & Cody, 1980; Cody, Reading & Line, 2003; Cody & Reading, 2005; Aiken et al., 2007; Saarela et al., 2013a; L. J. Gillespie & J. M. Saarela, 2016, unpublished data). Taxonomy follows Shultz (2006), who did not recognise infraspecific taxa, while Elven et al. (2011) recognised two: subsp. tilesii and subsp. elatior (Torr. & A. Gray) Hultén, but noted they may not be distinct.

Specimens Examined: Canada. Nunavut: Kitikmeot Region: Coppermine River, Fort Hearne–Bloody Falls [67.7761972°N, 115.2037222°W ± 7.5 km], 1931, A. M. Berry 28A (CAN-108352); Coppermine River, Fort Hearne–Bloody Falls [67.7761972°N, 115.2037222°W ± 7.5 km], 1931, A. M. Berry 28 (CAN-108365); Bloody Falls on Coppermine River, 67°44′N, 115°23′W, 27 July 1951, W. I. Findlay 201 (DAO-177628 01-01000616643); Coppermine [Kugluktuk], vic. of hamlet and airstrip, 67.78°N, 115.5°W ± 3,615 m, 23 June 1999, C. L. Parker & I. Jonsdottir 9101 (ALA); slopes on E side of Coppermine River, N of its confluence with Fockler Creek, ca. 0.8 km SW of Sandstone Rapids, 67°26′36.9″N, 115°38′50.1″W ± 50 m, 128 m, 4 July 2014, Saarela, Sokoloff & Bull 3390 (CAN, US); S side of Fockler Creek, ca. 2.7 SE of Sandstone Rapids, Coppermine River, 67°25′38.2″N, 115°36′54.9″W ± 50 m, 128 m, 5 July 2014, Saarela, Sokoloff & Bull 3406 (ALA, CAN, UBC); confluence of Coppermine River and Melville Creek, just W of Coppermine Mountains, 67°15′52″N, 115°30′55.3″W ± 350 m, 178–190 m, 7 July 2014, Saarela, Sokoloff & Bull 3490 (CAN, MO, MT, WIN); Kugluk (Bloody Falls) Territorial Park, sandy rocky beach at W side of Bloody Falls, 67°44′27.8″N, 115°22′20.3″W ± 25 m, 5 m, 13 July 2014, Saarela, Sokoloff & Bull 3913 (ALTA, CAN, O).

Askellia pygmaea (Ledeb.) Sennikov, Fig. 39—Dwarf alpine hawks-beard | Asian (C-NE)–amphi-Beringian–North American (NW) | Noteworthy Record

Figure 39 Askellia pygmaea.

(A) Habitat, Saarela et al. 3450. (B) Habit, Saarela et al. 3962. Photographs by R. D. Bull.

First report for the study area. We made collections at Fockler Creek and Kugluk (Bloody Falls) Territorial Park. It was recently reported from a site just west of the study area (Reading 586, DAO; Cody & Reading, 2005). These collections close a distribution gap between Tuktut Nogait Park and vicinity, southwestern Victoria Island, Bathurst Inlet and Hood River (Porsild & Cody, 1980; Gould & Walker, 1997; Cody & Reading, 2005; Saarela et al., 2013a). Elsewhere in the Canadian Arctic recorded from Baffin, Banks Melville, Prince Patrick, Southampton and Victoria islands, and scattered mainland sites (Porsild & Cody, 1980; Aiken et al., 2007; Saarela et al., 2013a). The species was previously treated in the genus Crepis L., as Crepis nana Richardson (Porsild & Cody, 1980; Bogler, 2006), described from one or more collections made along the Coppermine River between Point Lake and the coast (type/possible type K000808310, GH00006269), possibly within the study area (Richardson, 1823). It is now recognised in Askellia Weber (Enke, 2009; Elven et al., 2011), until recently as Askellia nana (Richardson) W. A. Weber. However, the epithet “pygmaea” has priority at species level when the taxon is included in Askellia, based on the synonym Prenanthes pygmaea Ledeb. (Chambers & Meyers, 2011). Chambers & Meyers (2011) proposed the combination Askellia pygmaea (Ledeb.) K. L. Chambers & S. C. Meyers, which is an isonym since the combination already existed (Sennikov & Illarionova, 2008).

Specimens Examined: Canada. Nunavut: Kitikmeot Region: E side of Fockler Creek, ridge above creek valley before its confluence with Coppermine River, ca. 1.8 km S of Sandstone Rapids, 67°26′3.9″N, 115°38′20.4″W ± 25 m, 168 m, 4 July 2014, Saarela, Sokoloff & Bull 3334 (CAN); N shore of Fockler Creek, ca. 2.3 km SSE of Sandstone Rapids, Coppermine River, 67°25′48″N, 115°37′33″W ± 25 m, 153 m, 5 July 2014, Saarela, Sokoloff & Bull 3450 (CAN, MO, UBC); S-facing sandstone cliffs above Coppermine River, ca. 7.8 km NNE of Sandstone Rapids, 67°31′15.1″N, 115°36′19.1″W ± 50 m, 106 m, 8 July 2014, Saarela, Sokoloff & Bull 3657 (CAN); Kugluk (Bloody Falls) Territorial Park, rocky beach above Bloody Falls, W bank of Coppermine River, 67°44′18″N, 115°22′57.3″W ± 250 m, 34 m, 14 July 2014, Saarela, Sokoloff & Bull 3962 (CAN, US); rocky beach along SE side of Coppermine River, above start of Bloody Falls, 67°44′16″N, 115°22′73.3″W ± 20 m, 17 m, 19 July 2014, Saarela, Sokoloff & Bull 4222 (CAN).

Erigeron eriocephalus J. Vahl—Woolly-headed fleabane | Circumpolar

Previously recorded from Kugluktuk (Cody, 1954b; Porsild & Cody, 1980). We made one collection just outside of Kugluktuk. Taxonomy follows Elven et al. (2011), who summarised taxonomic problems in the circumpolar-alpine “Erigeron uniflora aggregate”. The taxon has also been treated as Erigeron uniflorus var. eriocephalus (J. Vahl) B. Boivin (Nesom, 2006) and Erigeron uniflorus subsp. eriocephalus (J. Vahl) Cronquist (Cody, 2000). It was reported recently from a site just west of the study area (Reading 595, DAO; Cody & Reading, 2005). Elsewhere in the Canadian Arctic recorded from Axel Heiberg, Baffin, Banks, Coats, Melville, Prince Patrick, Southampton and Victoria islands, and scattered mainland sites (Porsild & Cody, 1980; Korol, 1992; Aiken et al., 2007; Saarela et al., 2013a).

Specimens Examined: Canada. Nunavut: Kitikmeot Region: Coppermine River, Fort Hearne–Bloody Falls [67.7761972°N, 115.2037222°W ± 7.5 km], 1931, A. M. Berry 22 (CAN-103793) & A. M. Berry 23 (CAN-103794); Coppermine [Kugluktuk], 67°49′36″N, 115°5′36″W, 24 July 1951, W. I. Findlay 178A (ACAD-30928, DAO-177056 01-01000617042, UBC-V40762); top of steep cliff overlooking Coppermine River, just S of Kugluktuk, 67°48′59″N, 115°6′15.6″W ± 50 m, 20 m, 26 July 2014, Saarela, Sokoloff & Bull 4422 (CAN).

Erigeron humilis Graham—Low fleabane | Amphi-Beringian–North American (N)–amphi-Atlantic (W)

Previously recorded from Kugluktuk (Cody, 1954b; Porsild & Cody, 1980). We made collections at Fockler Creek, Expeditor Cove, an island in the mouth of the Coppermine River, Kugluk (Bloody Falls) Territorial Park and Kugluktuk. Elsewhere in the Canadian Arctic recorded from Baffin, Banks, Coats, Southampton and Victoria islands, and scattered mainland sites (Porsild & Cody, 1980; Cody, Scotter & Zoltai, 1989; Korol, 1992; Aiken et al., 2007; Saarela et al., 2013a; Bennett, 2015). A third species of Erigeron, Erigeron compositus Pursh, was recorded for the study area in Porsild & Cody (1980), but we were unable to locate a voucher specimen. Pending confirmation we do not accept this record. Erigeron compositus reaches its known eastern limit on mainland Nunavut at Bathurst Inlet (Porsild & Cody, 1980; Cody, Scotter & Zoltai, 1984), but occurs much further east on some of the Arctic islands (Aiken et al., 2007).

Specimens Examined: Canada. Nunavut: Kitikmeot Region: Coppermine [Kugluktuk], 67°49′36″N, 115°5′36″W, 22 July 1951, W. I. Findlay 166 (DAO-177172 01-01000617044, MT00165029); Coppermine [Kugluktuk], near the settlement [67°49′36″N, 115°5′36″W ± 1.5 km], 3 July 1958, R. D. Wood s.n. (CAN-265444); old riverbed of Fockler Creek, ca. 2.3 km SSE of Sandstone Rapids, Coppermine River, 67°25′45.7″N, 115°37′21.8″W ± 25 m, 166 m, 2 July 2014, Saarela, Sokoloff & Bull 3170 (CAN); S side of Fockler Creek, ca. 2.7 SE of Sandstone Rapids, Coppermine River, 67°25′38.2″N, 115°36′54.9″W ± 50 m, 128 m, 5 July 2014, Saarela, Sokoloff & Bull 3403 (CAN, US); Coronation Gulf, NW peninsula of Expeditor Cove, ca. 9.5 km NW of Kugluktuk, 67°52′39.5″N, 115°16′43.8″W ± 10 m, 14 m, 8 July 2014, Saarela, Sokoloff & Bull 3703 (CAN, UBC); unnamed island just E (ca. 3.3 km) of Kugluktuk at mouth of Coppermine River, 67°49′29.2″N, 115°1′3.2″W ± 50 m, 1 m, 8 July 2014, Saarela, Sokoloff & Bull 3715 (CAN); flats atop and upper slopes of Coppermine Mountains, N/W side of Coppermine River, 67°14′49.9″N, 115°38′43.7″W ± 200 m, 467 m, 9 July 2014, Saarela, Sokoloff & Bull 3775 (CAN, MT); Kugluk (Bloody Falls) Territorial Park, SE-facing slope above small stream in deep gully that runs into Coppermine River just below Bloody Falls, ca. 1 km W of Bloody Falls, 67°44′41.2″N, 115°23′34.8″W ± 50 m, 49 m, 15 July 2014, Saarela, Sokoloff & Bull 4032 (ALA, CAN); Kugluk (Bloody Falls) Territorial Park, rocky valley immediately SW of Bloody Falls, along rough marked section of Portage Trail, 67°44′34″N, 115°22′16″W ± 50 m, 20 m, 18 July 2014, Saarela, Sokoloff & Bull 4171 (CAN); W of Kugluktuk on tundra flats above Coppermine River, S of 1 Coronation Drive and N of community power plant, 67°49′28.97″N, 115°5′0.2″W ± 100 m, 8 m, 22 July 2014, Saarela, Sokoloff & Bull 4250 (CAN, O); SE edge of Kugluktuk, rocky cliffs overlooking Coppermine River, 67°49′9.2″N, 115°5′40.4″W ± 50 m, 28 m, 24 July 2014, Saarela, Sokoloff & Bull 4352 (CAN).

Eurybia sibirica (L.) G. L. Nesom, Fig. 40—Arctic aster | European (N)–Asian (N/C)–amphi-Beringian–Cordilleran

Figure 40 Eurybia sibirica.

(A) Habitat, Saarela et al. 4113a. (B) Habit, Saarela et al. 4113a. Photographs by R. D. Bull.

Previously recorded from Kugluktuk (Cody, 1954b; Porsild & Cody, 1980). We made collections at Fockler Creek, Melville Creek, Kugluk (Bloody Falls) Territorial Park and other sites along the Coppermine River. This taxon was previously recognised in Aster L., as Aster sibiricus L. (Porsild & Cody, 1980; Nesom, 1994), but is now recognised in Eurybia (Cass.) Cass. (Brouillet, 2006a; Elven et al., 2011). Elven et al. (2011) recognised infraspecific taxa, with two reaching the Arctic, of which Eurybia sibirica var. gigantea (Spreng.) G. L. Nesom is the only one that reaches the Canadian Arctic (Elven et al., 2011). We follow Brouillet (2006a), who did not recognise infraspecific taxa. Elsewhere in the Canadian Arctic known from numerous mainland sites and not recorded from the Canadian Arctic Archipelago. The species is at the northern edge of its range in the study area and reaches its known eastern limit in North America at Bathurst Inlet, with no known collections from between there and the study area (Porsild & Cody, 1980; Saarela et al., 2013a; Bennett, 2015). Two collections (Reading 55, 584, DAO) reported in Cody & Reading (2005) from “sites midway between the south end of Bathurst [Inlet] and the vicinity of Coppermine [Kugluktuk]” are in fact from southwest of Kugluktuk. Several collections from the nearby Big Bend area of the Coppermine River were recently reported (Reading 21-1, 34, 34-1, 15, DAO; Cody, Reading & Line, 2003).

Several of our collections were found to be mixed with Symphyotrichum pygmaeum, a morphologically similar species. A key to distinguish Eurybia sibirica and Symphyotrichum pygmaeum is given in Saarela et al. (2013a). In addition to the characters listed there, leaf blade vestiture is helpful in distinguishing them (J. M. Saarela, 2016, personal observation), as described in the Flora of North America (Brouillet, 2006a; Brouillet et al., 2006): abaxial and adaxial blade surfaces sparsely woolly in Symphyotrichum pygmaeum vs. abaxial faces glabrescent to scabridulous, sparsely villous along veins, adaxial sparsely to ± densely villous or villoso-strigose in Eurybia sibirica. Blades in Symphyotrichum pygmaeum are generally narrower than those in Eurybia sibirica, particularly the distal ones.

Specimens Examined: Canada. Nunavut: Kitikmeot Region: Coppermine [Kugluktuk], 67°49′36″N, 115°5′36″W, 21 July 1951, W. I. Findlay 158 (DAO-176941 01-01000616981, MT00134713); Coppermine [Kugluktuk], near old Eskimo [sic] dwelling on Cemetery Island [67.834275°N, 115.0671833°W ± 0.8 km], 11 July 1958, R. D. Wood 93 (CAN-265448); Coppermine River, east bank, 10 July 1955, R. E. Miller 82 (CAN-242021); spruce forest along Fockler Creek, ca. 2.3 km SSE of Sandstone Rapids, Coppermine River, 67°25′45.7″N, 115°37′21.8″W ± 25 m, 166 m, 2 July 2014, Saarela, Sokoloff & Bull 3188 (CAN, US); slopes on E side of Coppermine River, N of its confluence with Fockler Creek, ca. 0.8 km SW of Sandstone Rapids, 67°26′36.9″N, 115°38′50.1″W ± 50 m, 128 m, 4 July 2014, Saarela, Sokoloff & Bull 3386b (CAN, O); S side of Fockler Creek, ca. 2.7 SE of Sandstone Rapids, Coppermine River, 67°25′38.2″N, 115°36′54.9″W ± 50 m, 128 m, 5 July 2014, Saarela, Sokoloff & Bull 3404 (ALA, ALTA, CAN); Coppermine River, sandstone cliffs above Sandstone Rapids, 67°27′29.6″N, 115°37′59.3″W ± 100 m, 110 m, 6 July 2014, Saarela, Sokoloff & Bull 3478b (CAN); Coppermine River, confluence of Coppermine River and Melville Creek, just W of Coppermine Mountains, 67°15′52″N, 115°30′55.3″W ± 350 m, 178–190 m, 7 July 2014, Saarela, Sokoloff & Bull 3489 (CAN, MO, O); S-facing sandstone cliffs above Coppermine River, ca. 7.8 km NNE of Sandstone Rapids, 67°31′15.1″N, 115°36′19.1″W ± 50 m, 106 m, 8 July 2014, Saarela, Sokoloff & Bull 3635a (CAN); Kugluk (Bloody Falls) Territorial Park, rocky beach above Bloody Falls, W bank of Coppermine River, 67°44′18″N, 115°22′57.3″W ± 250 m, 34 m, 14 July 2014, Saarela, Sokoloff & Bull 3964a (CAN, MT); Kugluk (Bloody Falls) Territorial Park, W side of Coppermine River, below Bloody Falls, 67°45′7.3″N, 115°22′20.1″W ± 3 m, 12 m, 17 July 2014, Saarela, Sokoloff & Bull 4137 (CAN, MT, UBC); Kugluk (Bloody Falls) Territorial Park, rocky sandy beach just below Bloody Falls, W side of Coppermine River, vicinity of confluence with small creek, beach seasonally flooded, 67°44′54.5″N, 115°22′17.2″W ± 75 m, 9 m, 17 July 2014, Saarela, Sokoloff & Bull 4113a (CAN, UBC); Kugluk (Bloody Falls) Territorial Park, S-facing slopes of large sand hill, NW of Bloody Falls rapids, 67°45′7.5″N, 115°22′43″W ± 3 m, 41 m, 18 July 2014, Saarela, Sokoloff & Bull 4172 (ALA, CAN); W of Kugluktuk on tundra flats above Coppermine River, S of 1 Coronation Drive and N of community power plant, 67°49′28.97″N, 115°5′0.2″W ± 100 m, 8 m, 22 July 2014, Saarela, Sokoloff & Bull 4258 (ALTA, CAN); grassy sand flats on extensive sandy floodplain of Coppermine River, below steep cliff above river and S of Kugluktuk, 67°48′54.3″N, 115°6′9.1″W ± 20 m, 5 m, 26 July 2014, Saarela, Sokoloff & Bull 4415b (CAN).

Hulteniella integrifolia (Richardson) Tzvelev, Fig. S19—Small arctic daisy | Amphi-Beringian–North American (N)

Previously recorded from Bloody Falls and Kugluktuk (Cody, 1954b; Porsild & Cody, 1980) and recently recorded from the nearby Big Bend area of the Coppermine River (Reading 9-1, 24, DAO; Cody, Reading & Line, 2003). Several Findlay collections (nos. 198, 199 and 239) at DAO were on loan and not available for study. We made collections at Fockler Creek, Coppermine Mountains, Kugluk (Bloody Falls) Territorial Park and Kugluktuk. Elsewhere in the Canadian Arctic recorded from Baffin, Banks, Coats, Devon, King William, Mansel, Mill, Prince of Wales, Somerset, Southampton and Victoria islands, and mainland sites (Porsild & Cody, 1980; Korol, 1992; Cody, 1996a; Cody, Reading & Line, 2003; Aiken et al., 2007; Saarela et al., 2013a). The basionym, Chrysanthemum integrifolium Richardson, was described from one or more collections made on the Copper Mountains [Coppermine Mountains] in the study area (type BM001025659, GH00004804, K000891745) (Richardson, 1823). It has also been recognised as Arctanthemum integrifolium (Richardson) Tzvelev (Bremer & Humphries, 1993), but is now placed in Hulteniella Tzvelev (Brouillet, 2006b; Elven et al., 2011). Inclusion of the monotypic Hulteniella in a recent molecular phylogenetic study found it to be a distinct lineage consistent with generic recognition (Riggins & Seigler, 2012).

Specimens Examined: Canada. Nunavut: Kitikmeot Region: Coppermine [Kugluktuk], Coronation Gulf, at mouth of Coppermine River, back of the village [67.822146°N, 115.078387°W ± 0.5 km], 4 August 1948, H. T. Shacklette 3288 (CAN-200096); Bloody Falls, 67°44′N, 115°23′W, 16 July 1951, W. I. Findlay 199 (MT00072107); E side of Fockler Creek, in valley just above creek’s confluence with the Coppermine River, ca. 1.4 km SSW of Sandstone Rapids, 67°26′14.5″N, 115°38′34.8″W ± 50 m, 146 m, 4 July 2014, Saarela, Sokoloff & Bull 3351 (CAN); forest and slopes at confluence of Big Creek and Coppermine River, N side of Coppermine River, S side of Coppermine Mountains, 67°14′29.3″N, 116°2′44.5″W ± 250 m, 180–199 m, 7 July 2014, Saarela, Sokoloff & Bull 3558 (CAN); Coronation Gulf, NW peninsula of Expeditor Cove, ca. 9.6 km NW of Kugluktuk, 67°52′39.1″N, 115°16′43.8″W ± 10 m, 25 m, 8 July 2014, Saarela, Sokoloff & Bull 3702 (CAN); Kugluk (Bloody Falls) Territorial Park, wet meadow between Coppermine River and large sand hills on W side of river, 0.5 km W of Bloody Falls, 67°44′44.8″N, 115°22′48.3″W ± 15 m, 33 m, 15 July 2014, Saarela, Sokoloff & Bull 4049 (CAN); Kugluk (Bloody Falls) Territorial Park, N-facing slope on high terrace above Bloody Falls rapids, SE side of Coppermine River, 67°44′16.2″N, 115°22′1.8″W ± 25 m, 91m, 19 July 2014, Saarela, Sokoloff & Bull 4226 (CAN, US); W of Kugluktuk on tundra flats above Coppermine River, S of 1 Coronation Drive and N of community power plant, 67°49′28.97″N, 115°5′0.2″W ± 100 m, 8 m, 22 July 2014, Saarela, Sokoloff & Bull 4248 (CAN, UBC).

Petasites frigidus subsp. sagittatus (Banks ex Pursh) Saarela, comb. et stat. nov. Basionym: Tussilago sagittata Banks ex Pursh, Flora Am. Sept. 531. 1814.—Arrow-leaved coltsfoot | North American | Noteworthy Record

First report of the taxon from the study area. Our single collection was made along Sleigh Creek. The plants were uncommon on moist ground in a dense willow thicket (Salix alaxensis) along the small creek, growing with Carex podocarpa, Chamerion angustifolium, Dasiphora fruticosa and Rubus arcticus subsp. acaulis. There are two recent unpublished collections from sites just outside the study area: one east of the confluence of the Coppermine and Kendall rivers (67°5′35″N, 115°43′20″W, 8 September 2002, Reading 599, DAO-788052 01-01000616638) and the other west of the Coppermine River (67°32′50″N, 116°13′0″W, 23 August 2002, Reading 596, DAO-788028 01-01000616637). All of these fill a distribution gap between northwestern Great Bear Lake and the Bathurst Inlet/Hood River area (Porsild, 1943; Porsild & Cody, 1980; Gould & Walker, 1997; Cherniawsky & Bayer, 1998c). This taxon has been reported from southeastern mainland Nunavut (Porsild, 1950b; Porsild & Cody, 1980; Cody, 1996a; Cody, Reading & Line, 2003; Cody & Reading, 2005), an area from which no records were included in the map in Cherniawsky & Bayer (1998c). It is not recorded for the Canadian Arctic Archipelago and is at the edge of its range in the study area. Cherniawsky & Bayer (1998c) and Bayer, Bogle & Cherniawsky (2006) recognised the taxon as Petasites frigidus var. sagittatus (Banks ex Pursh) Cherniawsky, while Porsild & Cody (1980) recognised it at species level as Petasites sagittatus Banks ex Pursh. Elven et al. (2011) tentatively accepted the taxon at species level, but treated the other taxa in the complex as subspecies: Petasites frigidus (L.) Fr. subsp. frigidus, Petasites frigidus subsp. nivalis (Greene) Cody, Petasites frigidus subsp. palmatus (Aiton) Cody and Petasites frigidus subsp. arcticus (A. E. Porsild) Cody. We follow the circumscriptions in Bayer, Bogle & Cherniawsky (2006), which are based on comprehensive revision of the genus in North America (Cherniawsky & Bayer, 1998a, 1998b, 1998c). To maintain consistency across the Arctic flora we prefer to recognise all of the taxa at subspecific rank, and the needed combination for subsp. sagittatus is here proposed.

Specimens Examined: Canada. Nunavut: Kitikmeot Region: Sleigh Creek, just downstream (W) of Tundra Lake, ca. 3.7 km SE of Sandstone Rapids, Coppermine River, 67°26′2.2″N, 115°33′42.9″W ± 25 m, 229 m, 5 July 2014, Saarela, Sokoloff & Bull 3449 (CAN, US).

Saussurea angustifolia (L.) DC. subsp. angustifolia, Fig. 41—Narrow-leaved sawwort | Amphi-Beringian (E)

Figure 41 Saussurea angustifolia subsp. angustifolia.

(A) Habit, Saarela et al. 3921. (B) Capitula, Kugluk (Bloody Falls) Territorial Park, Nunavut, 15 July 2014. Photographs by J. M. Saarela (A) and R. D. Bull (B).

Previously recorded from along the lower Coppermine River (Porsild & Cody, 1980). We made collections at Fockler Creek, Melville Creek, south of Escape Rapids, Kugluk (Bloody Falls) Territorial Park and Kugluktuk. This taxon extends eastward to the western shores of Hudson Bay but is not recorded for the Canadian Arctic Archipelago (Gussow, 1933; Porsild & Cody, 1980). It is at the northern edge of its range in the study area. Taxonomy follows Elven et al. (2011), who recognised two subspecies and one variety. Keil (2006) recognised the same three taxa, but all at the rank of variety. The subspecies angustifolia is the only taxon present in Arctic regions of Nunavut and Northwest Territories (Porsild & Cody, 1980). This distinctive, mauve-flowered species did not begin flowering until mid-July in the study area in 2014. Prior to flowering it is somewhat inconspicuous on the landscape.

Specimens Examined: Canada. Nunavut: Kitikmeot Region: Coppermine River, Fort Hearne–Bloody Falls [67.7761972°N, 115.2037222°W ± 7.5 km], 1931, A. M. Berry 27 (CAN-111005); Coppermine [Kugluktuk], 67°49′36″N, 115°5′36″W, 2 August 1995, T. Dolman 97 (LEA); S of Fockler Creek, along small tributary that runs into Fockler Creek, ca. 2.3 km S of Sandstone Rapids, Coppermine River, 67°25′44.9″N, 115°38′25.9″W ± 100 m, 152 m, 3 July 2014, Saarela, Sokoloff & Bull 3247 (CAN); confluence of Coppermine River and Melville Creek, just W of Coppermine Mountains, 67°15′52″N, 115°30′55.3″W ± 350 m, 178–190 m, 7 July 2014, Saarela, Sokoloff & Bull 3519 (CAN, US); S-facing slopes on W side of Coppermine River, about halfway between Escape Rapids and Muskox Rapids, 67°31′18.2″N, 115°36′20.1″W ± 150 m, 115 m, 8 July 2014, Saarela, Sokoloff & Bull 3619 (CAN); Kugluk (Bloody Falls) Territorial Park, flats above boardwalk W of Bloody Falls, 67°44′34.5″N, 115°22′27″W ± 100 m, 135 m, 13 July 2014, Saarela, Sokoloff & Bull 3921 (CAN, UBC); W of Kugluktuk on tundra flats above Coppermine River, S of 1 Coronation Drive and N of power plant, 67°49′28.97″N, 115°5′0.2″W ± 100 m, 8 m, 25 July 2014, Saarela, Sokoloff & Bull 4379 (CAN, O).

Senecio lugens Richardson, Fig. 42—Black-tipped groundsel | American Beringian–Cordilleran

Figure 42 Senecio lugens.

(A) Habit, Saarela et al. 3960. (B) Capitula, Saarela et al. 3192. (C) Habitat, Saarela et al. 3960. Photographs by R. D. Bull (A, C) and P. C. Sokoloff (B).

Previously recorded from Bloody Falls and Kugluktuk (Cody, 1954b; Porsild & Cody, 1980). We were unable to locate the voucher specimen from Bloody Falls (Findlay 148), but we do not doubt its identity. We made collections at Fockler Creek, Kugluk (Bloody Falls) Territorial Park and Kugluktuk. The species reaches its known eastern limit in the study area, where it is common and at the northern edge of its range. There are numerous mainland records from west of the study area (Porsild & Cody, 1980; Saarela et al., 2013a).

This species was described by Richardson in Franklin (1823: 747–748) from plants collected at Bloody Falls on 15 July 1821 (Houston, 1984). The protologue states “Hab. [Habitat] At Bloody Fall, where the Esquimaux were destroyed by the Northern Indians that accompanied Hearne, whence the specific name” (Richardson, 1823). The epithet lugens is the Latin word lugere that means mourning, grieving, or lamenting (https://en.wiktionary.org, accessed 11 August 2016). The type is housed at K (K000843626) and bears the text “Sea Coast Arctic Am [illegible] Dr. Richardson”, “Type” and includes the label “312 Senecio lugens”, corresponding to the number of the new taxon in Richardson’s Appendix to Franklin (1823). We interpret this to be the holotype. The species was not recorded for Nunavut in Barkley (2006) even though it was described from a location that is part of the territory, and collections are mapped from western Nunavut in Porsild & Cody (1980). Cameron (2015) discusses the naming of this species by Richardson, in relation to the events witnessed by Samuel Hearne at Bloody Falls in 1771.

Specimens Examined: Canada. Nunavut: Kitikmeot Region: Bloody Falls, near mouth of Coppermine River [67.742123°N, 115.370253°W ± 300 m], s.d., J. Richardson 312 (CAN-236062); Coppermine [Kugluktuk], 67°49′36″N, 115°5′36″W, 21 July 1951, W. I. Findlay 162 (DAO-178008, UBC-V229338); Coppermine [Kugluktuk], 67°49′36″N, 115°5′36″W, 2 August 1995, T. Dolman 100 (LEA); spruce forest along Fockler Creek, ca. 2.3 km SSE of Sandstone Rapids, Coppermine River, 67°25′45.7″N, 115°37′21.8″W ± 25 m, 166 m, 2 July 2014, Saarela, Sokoloff & Bull 3192 (CAN, US); slopes on E side of Coppermine River, N of its confluence with Fockler Creek, ca. 0.8 km SW of Sandstone Rapids, 67°26′36.9″N, 115°38′50.1″W ± 50 m, 128 m, 4 July 2014, Saarela, Sokoloff & Bull 3381 (CAN, UBC); S side of Fockler Creek, ca. 2.7 SE of Sandstone Rapids, Coppermine River, 67°25′38.2″N, 115°36′54.9″W ± 50 m, 128 m, 5 July 2014, Saarela, Sokoloff & Bull 3405 (ALA, ALTA, CAN, UBC); Kugluk (Bloody Falls) Territorial Park, rocky beach above Bloody Falls, W bank of Coppermine River, 67°44′18″N, 115°22′57.3″W ± 250 m, 34 m, 14 July 2014, Saarela, Sokoloff & Bull 3960 (CAN, O); top of steep cliff overlooking Coppermine River, just S of Kugluktuk, 67°48′59″N, 115°6′15.6″W ± 50 m, 20 m, 26 July 2014, Saarela, Sokoloff & Bull 4421 (ALTA, CAN).

Symphyotrichum pygmaeum (Lindl.) Brouillet & Selliah, Fig. 43—Pygmy aster | American Beringian

Figure 43 Symphyotrichum pygmaeum.

(A) Habit, Saarela et al. 4219. (B) Capitula, Saarela et al. 4219. Photographs by R. D. Bull.

Previously recorded from Kugluktuk (Cody, 1954b; Porsild & Cody, 1980). We made collections at Fockler Creek, Sandstone Rapids, Kugluk (Bloody Falls) Territorial Park and Kugluktuk. This species reaches its known eastern limit at the Hope Bay area northeast of Bathurst Inlet (Porsild & Cody, 1980; Cody & Reading, 2005). In the Canadian Arctic Archipelago recorded from Banks and Victoria islands (Aiken et al., 2007). It has been variously recognised as Aster pygmaeus Lindl. (Hultén, 1968; Porsild & Cody, 1980), Aster sibiricus subsp. pygmaeus (Lindl.) Á. Löve & D. Löve, Aster sibiricus var. pygmaeus (Lindl.) Cody and Eurybia pygmaeus (Lindl.) G. L. Nesom (Nesom, 1994), and was transferred to Symphyotrichum Nees on the basis of molecular evidence (Brouillet & Selliah, 2004). It is common along the upper, disturbed banks of the Coppermine River valley. There is a specimen from “mouth of the Coppermine [River]” in the Gray Herbarium (GH, not seen) (Cody, 1954b). Several of our collections were found to be mixed with Eurybia sibirica, a morphologically and ecologically similar species (see comments under that taxon).

Specimens Examined: Canada. Nunavut: Kitikmeot Region: Coppermine [Kugluktuk], 67°49′36″N, 115°5′36″W, 18 July 1951, W. I. Findlay 143 (DAO-176900 01-01000616980); slopes on E side of Coppermine River, N of its confluence with Fockler Creek, ca. 0.8 km SW of Sandstone Rapids, 67°26′36.9″N, 115°38′50.1″W ± 50 m, 128 m, 4 July 2014, Saarela, Sokoloff & Bull 3386a (CAN, US); Coppermine River, sandstone cliffs above Sandstone Rapids, 67°27′29.6″N, 115°37′59.3″W ± 100 m, 110 m, 6 July 2014, Saarela, Sokoloff & Bull 3478a (CAN); S-facing sandstone cliffs above Coppermine River, ca. 7.8 km NNE of Sandstone Rapids, 67°31′15.1″N, 115°36′19.1″W ± 50 m, 106 m, 8 July 2014, Saarela, Sokoloff & Bull 3635b (CAN); Kugluk (Bloody Falls) Territorial Park, rocky beach above Bloody Falls, W bank of Coppermine River, 67°44′18″N, 115°22′57.3″W ± 250 m, 34 m, 14 July 2014, Saarela, Sokoloff & Bull 3964b (CAN); Kugluk (Bloody Falls) Territorial Park, rocky sandy beach just below Bloody Falls, W side of Coppermine River, vicinity of confluence with small creek, beach seasonally flooded, 67°44′54.5″N, 115°22′17.2″W ± 75 m, 9 m, 17 July 2014, Saarela, Sokoloff & Bull 4113b (CAN); clay slopes and beach on E side of Coppermine River, just above start of Bloody Falls, 67°44′9.4″N, 115°22′41.2″W ± 15 m, 40 m, 19 July 2014, Saarela, Sokoloff & Bull 4219 (ALA, ALTA, CAN, MO, MT, O); grassy sandy flats on extensive sandy floodplain of Coppermine River, below steep cliff above river and S of Kugluktuk, 67°48′54.3″N, 115°6′9.1″W ± 20 m, 5 m, 26 July 2014, Saarela, Sokoloff & Bull 4415a (CAN).

Taraxacum ceratophorum (Ledeb.) DC., Fig. S20—Horned dandelion | Circumboreal-polar

Previously recorded from Kugluktuk, as Taraxacum lacerum Greene (Cody, 1954b; Porsild & Cody, 1980). We were unable to locate the voucher for one previous record (Findlay 132). We made numerous collections throughout the study area. Taxonomy follows Brouillet (2006c), who included the various taxa recognised in Porsild & Cody (1980)—Taraxacum lacerum, Taraxacum dumetorum Greene, Taraxacum pellianum A. E. Porsild, Taraxacum maurolepium G. E. Haglund, Taraxacum pseudonorvegicum Dahlstedt ex G. E. Haglund—in his concept of this species. This is the most common dandelion species in the study area, easily recognised by its conspicuously horned involucral bracts.

Specimens Examined: Canada. Nunavut: Kitikmeot Region: Coppermine [Kugluktuk], 67°49′36″N, 115°5′36″W, 3 July 1951, W. I. Findlay 83 (DAO-178167 01-01000616997); old riverbed of Fockler Creek, ca. 2.3 km SSE of Sandstone Rapids, Coppermine River, 67°25′48″N, 115°37′33″W ± 25 m, 153 m, 1 July 2014, Saarela, Sokoloff & Bull 3146 (CAN, US); meadow just S of Tundra Lake, ca. 4.2 km SE of Sandstone Rapids, Coppermine River, 67°25′34.8″N, 115°33′27.8″W ± 20 m, 265 m, 5 July 2014, Saarela, Sokoloff & Bull 3437 (CAN); W shore of Tundra Lake, ca. 4.3 km SE of Sandstone Rapids, Coppermine River, 67°25′43.7″N, 115°33′6.8″W ± 4 m, 262 m, 5 July 2014, Saarela, Sokoloff & Bull 3446 (CAN, UBC); forest and slopes at confluence of Big Creek and Coppermine River, N side of Coppermine River, S side of Coppermine Mountains, 67°14′29.3″N, 116°2′44.5″W ± 250 m, 180–199 m, 7 July 2014, Saarela, Sokoloff & Bull 3545 (CAN); confluence of Coppermine and Kendall rivers (NW side of Coppermine River, S side of Kendall River), 67°6′51.1″N, 116°8′18.3″W ± 150 m, 220 m, 7 July 2014, Saarela, Sokoloff & Bull 3576 (CAN); S-facing sandstone cliffs above Coppermine River, ca. 7.8 km NNE of Sandstone Rapids, 67°31′15.1″N, 115°36′19.1″W ± 50 m, 106 m, 8 July 2014, Saarela, Sokoloff & Bull 3636 (CAN); SE-facing slopes above Escape Rapids, W side of Coppermine River, 67°36′49.8″N, 115°29′27.4″W ± 10 m, 67 m, 8 July 2014, Saarela, Sokoloff & Bull 3738 (CAN, MT); Kugluk (Bloody Falls) Territorial Park, rocky cliffs and ledges directly above (W side) of Bloody Falls, just S of heavily used day-use/fishing area, 67°44′40.1″N, 115°22′4.9″W ± 20 m, 8 m, 12 July 2014, Saarela, Sokoloff & Bull 3822 (CAN, O); Kugluk (Bloody Falls) Territorial Park, upper ledges of rocky (gabbro) S-facing cliffs above the start of Bloody Falls (W bank of River), just E of Portage Trail, 67°44′21.7″N, 115°22′42.2″W ± 25 m, 46 m, 14 July 2014, Saarela, Sokoloff & Bull 3958 (ALTA, CAN); Kugluk (Bloody Falls) Territorial Park, rocky beach above Bloody Falls, W bank of Coppermine River, 67°44′18″N, 115°22′57.3″W ± 250 m, 34 m, 14 July 2014, Saarela, Sokoloff & Bull 3963 (CAN); SSW-facing slopes above start of Bloody Falls, SE side of Coppermine River, 67°44′12.5″N, 115°22′31″W ± 50 m, 50–60 m, 19 July 2014, Saarela, Sokoloff & Bull 4209 (CAN); rocky beach along SE side of Coppermine River, above start of Bloody Falls, 67°44′16″N, 115°22′73.3″W ± 20 m, 17 m, 19 July 2014, Saarela, Sokoloff & Bull 4223 (CAN); Kugluk (Bloody Falls) Territorial Park, day-use area above Bloody Falls (at outhouse and fire pit), 67°44′36.8″N, 115°22′11.1″W ± 25 m, 28 m, 20 July 2014, Saarela, Sokoloff & Bull 4234 (ALA, CAN);W of Kugluktuk on tundra flats above Coppermine River, S of 1 Coronation Drive and N of community power plant, 67°49′28.97″N, 115°5′0.2″W ± 100 m, 8 m, 22 July 2014, Saarela, Sokoloff & Bull 4257 (CAN); grassy vacant lot in Kugluktuk, 67°49′30.5″N, 115°5′29.3″W ± 15 m, 21 m, 24 July 2014, Saarela, Sokoloff & Bull 4342 (CAN, MO, QFA, WIN); SE edge of Kugluktuk, rocky cliffs overlooking Coppermine River, 67°49′9.2″N, 115°5′40.4″W ± 50 m, 28 m, 24 July 2014, Saarela, Sokoloff & Bull 4390 (CAN).

Taraxacum holmenianum Sahlin—Holmen’s dandelion | North American (N) | Noteworthy Record

Newly recorded for the study area. Our single collection was gathered at Richardson Bay on slightly higher ground above the estuary, where the species was uncommon growing with Alopecurus borealis, Androsace chamaejasme, Carex maritima and Salix niphoclada. There is an unpublished collection from a site west of the Coppermine River area, outside the study area, previously determined only to genus, which we have identified as this species (67°31′10″N, 116°10′0″W, 22 August 2002, Reading 589, DAO-788014 01-01000617005). A collection from Kugluktuk (Wood s.n.) was also re-determined to this species. The nearest collections are from Tuktut Nogait National Park and vicinity, southern Victoria Island and the Adelaide Peninsula (Porsild & Cody, 1980; Saarela et al., 2013a). Elsewhere in the Canadian Arctic recorded from Axel Heiberg, Baffin, Banks, Emerald, Melville, Prince of Wales, Prince Patrick, Ellesmere, Somerset and Victoria islands (Aiken et al., 2007). Recognised as Taraxacum pumilum Dahlstedt in Porsild & Cody (1980), a name now considered a synonym (Brouillet, 2006c).

Specimens Examined: Canada. Nunavut: Kitikmeot Region: Coppermine [Kugluktuk], edge of ditch behind Hudson’s Bay store [67°49′36″N, 115°5′36″W ± 1.5 km], 8 July 1958, R. D. Wood s.n. (CAN-253851); Richardson Bay, confluence of Richardson and Rae rivers at Coronation Gulf, ca. 20 km WNW of Kugluktuk, 67°54′11.2″N, 115°32′27.4″W ± 200 m, 0 m, 8 July 2014, Saarela, Sokoloff & Bull 3681 (CAN).

Taraxacum phymatocarpum J. Vahl—Northern dandelion | Circumpolar | Noteworthy Record

Previously recorded from Kugluktuk (Findlay 24) (Cody, 1954b; Porsild & Cody, 1980), but we were unable to find this voucher for confirmation. We made collections at Kugluktuk, Fockler Creek and Coppermine Mountains. The latter two sites represent a southern range extension for this Arctic species also recorded from Banks, Baffin, Devon, Emerald, Ellesmere, Melville, Prince Patrick and Victoria islands, and a few northern mainland sites (Porsild & Cody, 1980; Aiken et al., 2007).

Specimens Examined: Canada. Nunavut: Kitikmeot Region: slopes on E side of Coppermine River, N of its confluence with Fockler Creek, ca. 0.8 km SW of Sandstone Rapids, 67°26′36.9″N, 115°38′50.1″W ± 50 m, 128 m, 4 July 2014, Saarela, Sokoloff & Bull 3388 (CAN); meadow just S of Tundra Lake, ca. 4.2 km SE of Sandstone Rapids, Coppermine River, 67°25′34.8″N, 115°33′27.8″W ± 20 m, 265 m, 5 July 2014, Saarela, Sokoloff & Bull 3438 (CAN); flats atop and upper slopes of Coppermine Mountains, N/W side of Coppermine River, 67°14′49.9″N, 115°38′43.7″W ± 200 m, 467 m, 9 July 2014, Saarela, Sokoloff & Bull 3766 (CAN); W of Kugluktuk on tundra flats above Coppermine River, S of 1 Coronation Drive and N of community power plant, 67°49′28.97″N, 115°5′0.2″W ± 100 m, 8 m, 22 July 2014, Saarela, Sokoloff & Bull 4238 (CAN).

Tephroseris frigida (Richardson) Holub—Arctic groundsel | Amphi-Beringian–North American (NW)

Previously recorded from Kugluktuk, as Senecio frigidus (Richardson) Less. (Cody, 1954b) and Senecio atropurpureus (Ledeb.) B. Fedtsch. (Porsild & Cody, 1980), but now recognised in Tephroseris (Rchb.) Rchb. (Barkley & Murray, 2006; Elven et al., 2011). We made collections at Fockler Creek and Kugluk (Bloody Falls) Territorial Park. Elsewhere in the Canadian Arctic known from Banks and Victoria islands and mainland sites (Porsild & Cody, 1980; Aiken et al., 2007; Saarela et al., 2013a), reaching its known eastern limit in the Bathurst Inlet area (Porsild & Cody, 1980; Bennett, 2015). The basionym, Cineraria frigida Richardson, was described from one or more collections made by J. Richardson along the Coppermine River between Point Lake and the Arctic coast, possibly within the study area (type BM001041629, BM001041630).

Specimens Examined: Canada. Nunavut: Kitikmeot Region: Coppermine [Kugluktuk], 67°49′36″N, 115°5′36″W, 24 July 1951, W. I. Findlay 173 (DAO-177812 01-01000616651); Coppermine [Kugluktuk], 67°49′36″N, 115°5′36″W, 1 August 1951, W. I. Findlay 231 (ACAD-30923, DAO-177810 01-01000616652, UBC-V40754); Rocky Defile [Rapids], on the Coppermine River, 25 July 1995, T. Dolman 99 (LEA); NW-facing slope just upstream of small tributary from its confluence with Fockler Creek, ca. 2.4 km SSW of Sandstone Rapids, Coppermine River, 67°25′46″N, 115°38′49.4″W ± 200 m, 149 m, 3 July 2014, Saarela, Sokoloff & Bull 3309 (CAN, US); Kugluk (Bloody Falls) Territorial Park, flats above boardwalk W of Bloody Falls, 67°44′34.5″N, 115°22′27″W ± 100 m, 135 m, 13 July 2014, Saarela, Sokoloff & Bull 3918 (CAN, MT); Kugluk (Bloody Falls) Territorial Park, rocky valley immediately SW of Bloody Falls, along rough marked section of Portage Trail, 67°44′34″N, 115°22′16″W ± 50 m, 20 m, 18 July 2014, Saarela, Sokoloff & Bull 4163 (CAN).

Tephroseris palustris subsp. congesta (R. Br.) Holub, Fig. 44—Marsh groundsel | European (C-S) & European (NE)–Asian (N/C)–amphi-Beringian–North American

Figure 44 Tephroseris palustris subsp. congesta.

(A) Habitat, Kugluktuk, Nunavut, 23 July 2014. (B) Capitula, Saarela et al. 4344. Photographs by R. D. Bull.

Previously recorded from Kugluktuk, as Senecio congestus R. Br. (Cody, 1954b; Porsild & Cody, 1980) and now recognised in Tephroseris (Barkley & Murray, 2006; Elven et al., 2011). We made collections in Kugluktuk and on an island in the mouth of the Coppermine River. Recorded elsewhere in the Canadian Arctic from numerous Arctic and mainland sites (Porsild & Cody, 1980; Korol, 1992; Cody & Reading, 2005; Aiken et al., 2007).

Specimens Examined: Canada. Nunavut: Kitikmeot Region: Coppermine [Kugluktuk], 67°49′36″N, 115°5′36″W, 20 July 1951, W. I. Findlay 157 (DAO-177873 01-01000617002, UBC-V40756); Coppermine [Kugluktuk], 67°51′N, 115°16′W, 2 July 1972, F. Fodor N 164 (UBC-V151899); Coppermine [Kugluktuk], 67°49′36″N, 115°5′36″W, 2 August 1995, T. Dolman 101 (LEA); Kugluktuk, corner of Tuktu Road and Saddleback Street, 67°49′23.8″N, 115°6′17.6″W ± 10 m, 21 m, 29 June 2014, Saarela, Sokoloff & Bull 3093 (CAN); unnamed island just E (ca. 3.3 km) of Kugluktuk at mouth of Coppermine River, 67°49′29.2″N, 115°1′3.2″W ± 50 m, 1 m, 8 July 2014, Saarela, Sokoloff & Bull 3725 (CAN); grassy vacant lot in Kugluktuk, 67°49′30.5″N, 115°5′29.3″W ± 15 m, 21 m, 24 July 2014, Saarela, Sokoloff & Bull 4344 (CAN, UBC, US).

Tripleurospermum maritimum subsp. phaeocephalum (Rupr.) Hämet-Ahti—Arctic chamomile | Circumpolar

Previously recorded for the study area, as Matricaria ambigua (Ledeb.) Krylov (Porsild & Cody, 1980), based on a collection from Richardson Bay. We found this seashore species only once, in Kugluktuk. Elsewhere in the Canadian Arctic recorded from Baffin, Banks, Southampton and Victoria islands, and mainland sites (Porsild & Cody, 1980; Cody, Scotter & Zoltai, 1989; Korol, 1992; Cody & Reading, 2005; Aiken et al., 2007). Taxonomy follows Elven et al. (2011) and Brouillet (2006d), who accept three subspecies. Two are present in Canada, but only this one occurs in the Canadian Arctic.

Specimens Examined: Canada. Nunavut: Kitikmeot Region: Coppermine [Kugluktuk], field near beach, south shore of Richardson Bay, 7 July 1955, R. E. Miller 40 (CAN-242025); W of Kugluktuk on tundra flats above Coppermine River, S of 1 Coronation Drive and N of community power plant, 67°49′28.97″N, 115°5′0.2″W ± 100 m, 8 m, 26 July 2014, Saarela, Sokoloff & Bull 4424 (CAN).

Betulaceae [2/3]

Alnus alnobetula (Ehrh.) K. Koch, Fig. 45—American green alder | European (C) & European (NE)–Asian (N/C)–amphi-Beringian–North American | Noteworthy Collection

Figure 45 Alnus alnobetula.

(A) Staminate and pistillate catkins, Saarela et al. 3598. (B) Habit, Saarela et al. 3598. Photographs by P. C. Sokoloff.

Newly recorded for the study area. The shrubs formed a dense thicket along the upper banks of Bigtree River, a tributary running into the Coppermine River, in an area of dense white spruce forest, associated with Betula glandulosa, Bromus pumpellianus, Dasiphora fruticosa and Salix alaxensis. This Subarctic collection site is slightly beyond the range for the species given in Furlow (1997). The taxon does not reach the Arctic in the study area, but has been recorded from the Arctic ecozone in Nunavut at Bathurst Inlet (Cody, 1954a; Porsild & Cody, 1980; Bennett, 2015), where the shrubs were reported as growing up to five feet high, and along the upper Hood River (Gould & Walker, 1997). Bennett (2015) noted observations of the species from a helicopter upriver of Wilberforce Falls on the Hood River. Elsewhere in Nunavut recorded from the southeastern mainland (Porsild, 1950b; Porsild & Cody, 1980; Cody, Reading & Line, 2003). It also reaches the Arctic in the Mackenzie Delta area and northern Quebec and Labrador (Porsild & Cody, 1980; Garneau, 2015a).

Alnus alnobetula is the correct name for the taxon long recognised as Alnus viridis (Chaix) DC. (Greuter & Raab-Straube, 2011; Chery, 2015). Recognition of taxa in the circumpolar Alnus viridis complex is problematic (Banaev & Adel’shin, 2009). In a recent treatment, Furlow (1997) recognised three subspecies in North America. He recognised plants from central and eastern North America and Greenland as Alnus viridis subsp. crispa (=Alnus alnobetula subsp. crispa (Aiton) Raus), plants from northwestern North America as subsp. fruticosa (Ruprecht) Nyman (=Alnus alnobetula subsp. fruticosa (Rupr.) Raus), and plants from western North America as subsp. sinuata (Regel) Á. Löve & D. Löve (=Alnus alnobetula subsp. sinuata (Regel) Raus). In his earlier monograph of American Alnus, Furlow (1979) treated Alnus viridis subsp. crispa as ranging across Canada (as did Porsild & Cody, 1980), and considered subsp. fruticosa as a synonym of that taxon. In his more recent treatment (Furlow, 1997), the ranges of subspecies crispa and fruticosa are shown to overlap in Northwest Territories and Nunavut in the area north of the Manitoba and Alberta border, and both are mapped as occurring at Bathurst Inlet. There is some confusion in this treatment, however, as the distributions listed in the key to the three Alnus viridis subspecies are discordant with the distributions listed after the taxon descriptions. Banaev & Adel’shin (2009) provide a specimen-based map of the North American distributions of Alnus viridis subsp. crispa and subsp. fruticosa, which they note to be “according to Furlow (1997, FNA treatment)”—but it is unclear how they arrived at the identifications of the specimens on which the distributions are based, particularly those in the zone of overlap of the two subspecies, as (1) Furlow (1997) included only range maps, not dot maps; (2) we are not aware of any other specimen-based map for these taxa; and (3) none of the specimens at DAO and CAN, where the majority of specimens from these territories are housed, have been annotated as one or the other of these taxa. The subspecies as recognised by Furlow (1997) are extremely difficult to distinguish (J. M. Saarela, 2016, personal observation) and for this reason we refrain from assigning our new collection to an infraspecific taxon, pending further study of the variation in northern North America.

Specimens Examined: Canada. Nunavut: Kitikmeot Region: confluence of Coppermine and Bigtree rivers, 66°56′23.8″N, 116°21′3.2″W ± 100 m, 265 m, 7 July 2014, Saarela, Sokoloff & Bull 3598 (ALA, CAN, UBC).

Betula glandulosa Michx.—Dwarf birch | North American (N)

Previously recorded from Bloody Falls and Kugluktuk (Cody, 1954b; Porsild & Cody, 1980). We were unable to locate some earlier collections from Bloody Falls (Findlay 138) and Kugluktuk (Findlay 47, 64) for confirmation, but it is unlikely these sheets of this well-known species are misidentified. We made collections at Fockler Creek, Melville Creek, Kugluk (Bloody Falls) Territorial Park and Heart Lake. Elsewhere in the Canadian Arctic recorded from southern Baffin Island, Banks, Southampton and Victoria islands, and across the mainland (Porsild & Cody, 1980; Cody, Scotter & Zoltai, 1989; Korol, 1992; Cody, 1996a; Aiken et al., 2007; Saarela et al., 2013a; Garneau, 2015a).

Betula glandulosa and Betula nana subsp. exilis (Sukaczev) Hultén are northern-ranging taxa that have been variously recognised as a single species, due to considerable morphological intergradation where their ranges overlap (Hultén, 1968), or separate taxa (reviewed in de Groot, Thomas & Wein, 1997). For example, Furlow (1997), Elven et al. (2011), Cody (2000), and Hultén (1968) treat them as separate taxa, whereas Porsild & Cody (1980) apparently treat them as a single taxon, since the name Betula nana is not mentioned. According to the maps in Furlow (1997), the ranges of both taxa overlap extensively on mainland Nunavut and Northwest Territories, including in the study area. Although numerous molecular studies of Betula have been conducted (Järvinen et al., 2004; Li, Shoup & Chen, 2007; Schenk et al., 2008), Betula nana and Betula glandulosa have been included together in only a single molecular study based on nuclear ribosomal ITS sequences, in which Betula nana (subsp. nana; an accession from Sweden) and Betula glandulosa were part of different clades (Li, Shoup & Chen, 2005); their phylogenetic affinities are unclear. Populations of Betula glandulosa from the Canadian Arctic are genetically distinct compared to those from Greenland, Iceland and Europe (Alsos et al., 2015).

Using the key in Furlow (1997), our collections key to Betula nana subsp. exilis; however, there is a problem in the key. Although there is no overlap in the ranges given for leaf length in lead 10 separating the taxa into groups [0.5–2 cm for a group of taxa including Betula nana and 2.5–5(–7) cm for a group of taxa including Betula glandulosa], the subsequent lead to Betula glandulosa (lead 14+) contradicts the earlier one (lead 10+) by including blades that are up to one cm shorter than stated in the earlier lead [leaf blades 1–2(–4) cm]. Moreover, the leaf blade length range given in the description for Betula glandulosa (0.5–3 cm) differs from that in the final lead to the taxon, and it fully overlaps with the range given in the description of Betula nana subsp. exilis (0.5–1.2 cm). The circumscriptions of these taxa are unclear, and a detailed taxonomic study combining morphological and molecular characters is needed. Pending clarification of the status of these taxa, we treat them as a single taxon under Betula glandulosa, as in Porsild & Cody (1980), even though the name Betula nana has priority.

Specimens Examined: Canada. Nunavut: Kitikmeot Region: Coppermine [Kugluktuk], vicinity of post [67°49′36″N, 115°5′36″W ± 1.5 km], 26 July 1949, A. E. Porsild 17170 (CAN-127669); Kugluktuk, 67.81875°N, 115.086833°W, 10 July 2006, J. Davis 623 (CAN-597648); Kugluktuk, rocky slopes of North Hill, 67°49′29.6″N, 115°6′31″W ± 50 m, 50 m, 29 June 2014, Saarela, Sokoloff & Bull 3044 (ALA, ALTA, CAN, MO, UBC); flats on W side of Fockler Creek, above spruce forest in creek valley, ca. 2.2 km S of Sandstone Rapids, Coppermine River, 67°25′49″N, 115°37′55″W ± 50 m, 152 m, 1July 2014, Saarela, Sokoloff & Bull 3132 (CAN, MO, MT, O); confluence of Coppermine River and Melville Creek, just W of Coppermine Mountains, 67°15′52″N, 115°30′55.3″W ± 350 m, 178–190 m, 7 July 2014, Saarela, Sokoloff & Bull 3525 (CAN, NY, QFA, WIN); S-facing slopes above Coppermine River, ca. 7.8 km NNE of Sandstone Rapids, 67°31′16.2″N, 115°36′52.1″W ± 50 m, 110 m, 8 July 2014, Saarela, Sokoloff & Bull 3644 (CAN, UBC); Kugluk (Bloody Falls) Territorial Park, rocky cliffs and ledges directly above (W side) of Bloody Falls, just S of heavily used day-use/fishing area, 67°44′40.1″N, 115°22′4.9″W ± 20 m, 8 m, 12 July 2014, Saarela, Sokoloff & Bull 3826 (ALA, ALTA, CAN); Kugluk (Bloody Falls) Territorial Park, W side of Coppermine River, just above Bloody Falls, 67°44′22.6″N, 115°22′52″W ± 20 m, 40 m, 16 July 2014, Saarela, Sokoloff & Bull 4106 (CAN, US); ca. 0.5 km SW of Heart Lake, SW of Kugluktuk, 7.5 km SW of mouth of Coppermine River, 67°47′52″N, 115°14′14.4″W ± 350 m, 66 m, 23 July 2014, Saarela, Sokoloff & Bull 4280 (CAN).

Betula occidentalis Hook., Fig. 46—Water birch | North American | Noteworthy Record

Figure 46 Betula occidentalis.

(A) Pistillate catkins, Saarela et al. 4079. (B) Habit, Saarela et al. 4318. Photographs by R. D. Bull.

Newly recorded for the study area, and a range extension from the nearest known populations from the eastern shore of Great Bear Lake (Dugle, 1966; Porsild & Cody, 1980; Furlow, 1997). We found this species at eight sites throughout the area and as far north as Kugluktuk, growing on south-facing slopes, usually on or near upper ridges, as a small, multi-branched shrub from two to 13 feet tall, and always larger than the Betula glandulosa shrubs amongst which it grew. At each collection site only one to three plants were found, similar to the frequency for the species noted elsewhere at a northern site (Scott, Staniforth & Fayle, 1992). Associated species include Betula glandulosa, Dasiphora fruticosa, Empetrum nigrum, Juniperus communis subsp. depressa, Linnaea borealis subsp. americana, Picea glauca, Pyrola grandiflora, Rhododendron groenlandicum, Salix reticulata, Vaccinium uliginosum and Vaccinium vitis-idaea. It was recently observed (July 2016) in the study area about halfway between Kugluktuk and Richardson Bay (67.8513°N, 115.3262°W) (M. Lamont, 2016, personal communication and photo!, confirmed by J. M. Saarela). It also reaches the Arctic in the Mackenzie Delta area (Porsild & Cody, 1980; Furlow, 1997), Bathurst Inlet (Scotter and Zoltai 31969, DAO; Cody, Scotter & Zoltai, 1984) and Burnside River near Bathurst Inlet (Bennett, 2015), but has not been reported from sites between these areas. The earlier Bathurst Inlet record is neither mapped in Furlow (1997) nor is the species included in the Checklist of the Panarctic Flora (Elven et al., 2011). Elsewhere in Nunavut recorded on the southeastern mainland (Porsild, 1950b; Porsild & Cody, 1980). In the portion of Furlow’s (1997) key distinguishing Betula occidentalis from Betula kenaica W. H. Evans (Alaska, Yukon) and Betula minor (Tuck.) Fernald (northeastern North America), the central lobe of infructescence scales is described as being “much shorter than lateral lobes”, whereas in the description it is described as “narrower and longer than ascending lateral lobes”; the latter is correct.

Specimens Examined: Canada. Nunavut: Kitikmeot Region: confluence of Coppermine and Kendall rivers (NW side of Coppermine River, S side of Kendall River), 67°6′51.1″N, 116°8′18.3″W ± 150 m, 220 m, 7 July 2014, Saarela, Sokoloff & Bull 3572 (ALA, ALTA, CAN, MO, MT, QFA); confluence of Coppermine and Bigtree rivers, 66°56′23.8″N, 116°21′3.2″W ± 100 m, 265 m, 7 July 2014, Saarela, Sokoloff & Bull 3604 (CAN, MO, MT); S-facing slopes above Coppermine River, ca. 7.8 km NNE of Sandstone Rapids, 67°31′16.2″N, 115°36′52.1″W ± 50 m, 110 m, 8 July 2014, Saarela, Sokoloff & Bull 3645 (ALA, CAN); Kugluk (Bloody Falls) Territorial Park, S-facing cliff (gabbro sill) above start of Bloody Falls, W side of Coppermine River, W side of Portage Trail, 67°44′23.2″N, 115°22′54.5″W ± 50 m, 57 m, 16 July 2014, Saarela, Sokoloff & Bull 4079 (CAN, O, UBC, US, WIN); SSW-facing slopes above start of Bloody Falls, SE side of Coppermine River, 67°44′12.5″N, 115°22′31″W ± 50 m, 50–60 m, 19 July 2014, Saarela, Sokoloff & Bull 4211 (ALTA, CAN); N side of Heart Lake, below rocky cliff, SW of Kugluktuk, 5.64 km SW of mouth of Coppermine River, 67°48′33.8″N, 115°12′52.9″W ± 15 m, 39 m, 23 July 2014, Saarela, Sokoloff & Bull 4318 (CAN, NY, QFA, WIN); rocky cliffs on S side of Kugluktuk, 67°49′13″N, 115°5′55.8″W ± 50 m, 65 m, 26 July 2014, Saarela, Sokoloff & Bull 4405 (CAN, O); top of steep cliff overlooking Coppermine River, just S of Kugluktuk, 67°48′59″N, 115°6′15.6″W ± 50 m, 20 m, 26 July 2014, Saarela, Sokoloff & Bull 4420 (CAN, UBC, US).

Boraginaceae [1/1]

Mertensia maritima subsp. tenella (Th. Fr.) Elven & Skarpaas, Fig. 47A—Seaside bluebells, sea-lungwort | Amphi-Beringian–North American (N)–amphi-Atlantic (W)

Figure 47 Mertensia maritima subsp. tenella and Braya humilis subsp. humilis.

Mertensia maritima subsp. tenella: (A) habit, Saarela et al. 3714. Braya humilis subsp. humilis: (B) habit, Saarela et al. 3965. Photographs by P. C. Sokoloff.

Previously recorded from Kugluktuk (Cody, 1954b; Porsild & Cody, 1980). We found this distinctive coastal species only on an island in the mouth of the Coppermine River. Elsewhere in the Canadian Arctic recorded from Baffin, Banks, Devon, Coats, Southampton and Victoria islands, and several mainland sites (Porsild & Cody, 1980; Cody, Scotter & Zoltai, 1989; Korol, 1992; Aiken et al., 2007; Saarela et al., 2013a).

Specimens Examined: Canada. Nunavut: Kitikmeot Region: Coppermine [Kugluktuk], 67°49′36″N, 115°5′36″W, 24 July 1951, W. I. Findlay 176 (DAO-178958 01-01000619666, UBC-V40800); Coppermine [Kugluktuk], 67°49′36″N, 115°5′36″W, 1 July 1951, W. I. Findlay 58 (DAO-832282 01-01000619665); unnamed island just E (ca. 3.3 km) of Kugluktuk at mouth of Coppermine River, 67°49′29.2″N, 115°1′3.2″W ± 50 m, 1 m, 8 July 2014, Saarela, Sokoloff & Bull 3714 (CAN).

Brassicaceae [8/18]

Braya glabella Richardson subsp. glabella—Smooth northern rockcress | Amphi-Beringian–North American (N)

We collected this species at two sites at Fockler Creek, where it grew in a wet sedge meadow and mesic to wet tundra. Although described from the Copper Mountains [Coppermine Mountains] (isotype GH0018822!) within the study area, the type location was not mapped in contemporary treatments, which show Bathurst Inlet and eastern Great Bear Lake as the nearest locations (Porsild & Cody, 1980; Harris, 1985). Elsewhere in the Canadian Arctic recorded from Baffin, Banks, Southampton and Victoria islands, and a few mainland sites (Porsild & Cody, 1980; Cody & Reading, 2005; Aiken et al., 2007; Saarela et al., 2013a).

Specimens Examined: Canada. Nunavut: Kitikmeot Region: sedge meadow adjacent to small lake on flats N of Fockler Creek, ca. 1.5 km SSE of Sandstone Rapids, Coppermine River, 67°26′8.8″N, 115°37′35.9″W ± 20 m, 168 m, 2 July 2014, Saarela, Sokoloff & Bull 3225 (CAN); flats N of Fockler Creek, between Fockler Creek and Sleigh Creek, ca. 0.6 km SE of Sandstone Rapids, Coppermine River, 67°26′43.3″N, 115°37′40″W ± 3 m, 157 m, 6 July 2014, Saarela, Sokoloff & Bull 3484 (ALA, CAN, UBC).

Braya humilis (C. A. Mey.) B. L. Rob. subsp. humilis, Fig. 47B—Low rockcress, Northern rockcress | Asian (N/C)–amphi-Beringian–North American (N) | Noteworthy Record

Newly recorded for the study area. We made collections at Fockler Creek, Melville Creek, Big Creek and Kugluk (Bloody Falls) Territorial Park, which close a distribution gap between eastern Great Bear Lake, Tuktut Nogait National Park and vicinity, Banks and Victoria islands, and the Bathurst Inlet area (Porsild & Cody, 1980; Harris, 1985; Cody & Reading, 2005; Saarela et al., 2013a); the latter is the known eastern limit of the taxon on mainland Nunavut. Taxonomy follows Harris (2006, 2010), whereas Al-Shehbaz & German (2014) do not recognise infraspecific taxa. This taxon is recognised as Braya richardsonii (Rydb.) Fernald in Porsild & Cody (1980).

Specimens Examined: Canada. Nunavut: Kitikmeot Region: slopes on E side of Coppermine River, N of its confluence with Fockler Creek, ca. 0.8 km SW of Sandstone Rapids, 67°26′36.9″N, 115°38′50.1″W ± 50 m, 128 m, 4 July 2014, Saarela, Sokoloff & Bull 3397 (CAN, UBC); confluence of Coppermine River and Melville Creek, just W of Coppermine Mountains, 67°15′52″N, 115°30′55.3″W ± 350 m, 178–190 m, 7 July 2014, Saarela, Sokoloff & Bull 3493 (CAN); forest and slopes at confluence of Big Creek and Coppermine River, N side of Coppermine River, S side of Coppermine Mountains, 67°14′29.3″N, 116°2′44.5″W ± 250 m, 180–199 m, 7 July 2014, Saarela, Sokoloff & Bull 3562 (CAN); SE-facing slopes above Escape Rapids, W side of Coppermine River, 67°36′49.8″N, 115°29′27.4″W ± 10 m, 67 m, 8 July 2014, Saarela, Sokoloff & Bull 3737 (ALTA, CAN); Kugluk (Bloody Falls) Territorial Park, rocky valley immediately SW of Bloody Falls, along rough marked section of Portage Trail, 67°44′34″N, 115°22′16″W ± 50 m, 20 m, 13 July 2014, Saarela, Sokoloff & Bull 3860 (CAN); Kugluk (Bloody Falls) Territorial Park, rocky valley immediately SW of Bloody Falls, along rough marked section of Portage Trail, upper pond just W of Bloody Falls, 67°44′39.5″N, 115°22′28.9″W ± 10 m, 15 m, 13 July 2014, Saarela, Sokoloff & Bull 3897 (CAN); Kugluk (Bloody Falls) Territorial Park, rocky beach above Bloody Falls, W bank of Coppermine River, 67°44′18″N, 115°22′57.3″W ± 250 m, 34 m, 14 July 2014, Saarela, Sokoloff & Bull 3965 (ALA, CAN); Kugluk (Bloody Falls) Territorial Park, rocky sandy beach just below Bloody Falls, W side of Coppermine River, vicinity of confluence with small creek, beach seasonally flooded, 67°44′54.5″N, 115°22′17.2″W ± 75 m, 9 m, 17 July 2014, Saarela, Sokoloff & Bull 4114 (CAN).

Cardamine bellidifolia L.—Alpine bittercress | Circumpolar-alpine | Noteworthy Record

Our two collections from Fockler Creek are the first records for the study area. It was uncommon along the lower edge of a snowbed community on a north-facing slope, growing with Draba simmonsii, Ranunculus nivalis, Ranunculus pygmaeus and the lichen Masonhalea richardsonii (Hook.) Kärnefelt. There is a nearby record from a site east of the confluence of the Kendall and Coppermine rivers (Reading 302, DAO; Cody & Reading, 2005). These collections close a distribution gap between Bathurst Inlet, Hood River and the Mackenzie Delta area (Porsild & Cody, 1980; Gould & Walker, 1997). It is conspicuously absent from most of Banks and Victoria islands, but known from many sites throughout the rest of the Canadian Arctic Archipelago and several sites on the Arctic mainland (Porsild & Cody, 1980; Cody, Scotter & Zoltai, 1989; Korol, 1992; Cody & Reading, 2005; Aiken et al., 2007). Taxonomy follows Al-Shehbaz, Marhold & Lihová (2010) and Elven et al. (2011); the latter recognised the taxon as subsp. bellidifolia with no mention of other Arctic subspecies.

Specimens Examined: Canada. Nunavut: Kitikmeot Region: flats on W side of Fockler Creek, above spruce forest in creek valley, ca. 2.2 km S of Sandstone Rapids, Coppermine River, 67°25′49″N, 115°37′55″W ± 50 m, 152 m, 1 July 2014, Saarela, Sokoloff & Bull 3140 (CAN); old riverbed of Fockler Creek, ca. 2.3 km SSE of Sandstone Rapids, Coppermine River, 67°25′45.7″N, 115°37′21.8″W ± 25 m, 166 m, 2 July 2014, Saarela, Sokoloff & Bull 3180 (CAN).

Cardamine digitata Richardson—Richardson’s bittercress | Amphi-Beringian-North American (NW)

Previously recorded from Bloody Falls and Kugluktuk (Cody, 1954b; Porsild & Cody, 1980). We were unable to locate Findlay 135 (DAO-184915) from Bloody Falls for confirmation. We made collections in these same areas, and at Fockler Creek and Heart Lake. This species was not recorded for Nunavut in Al-Shehbaz, Marhold & Lihová (2010) although it is widespread on mainland Nunavut (Porsild & Cody, 1980; Korol, 1992). Elsewhere in the Canadian Arctic recorded from Banks, Prince Charles, Southampton and Victoria islands (Aiken et al., 2007). It was described by Richardson (1823) from one or more collections made along the Coppermine River from Point Lake to the coast, possibly within the study area. There is putative original material at BM (BM000584314), K (K000697786) and GH (GH000189097).

Specimens Examined: Canada. Nunavut: Kitikmeot Region: Coppermine [Kugluktuk] [67°49′36″N, 115°5′36″W ± 1.5 km], 6 July 1958, R. D. Wood s.n. (CAN-265542); Coppermine [Kugluktuk], 67°49′36″N, 115°5′36″W, 2 July 1951, W. I. Findlay 61 (ALTA-VP-9553, DAO-184914, UBC-V40793); Kugluktuk, rocky slopes of North Hill, 67°49′29.6″N, 115°6′31″W ± 50 m, 50 m, 29 June 2014, Saarela, Sokoloff & Bull 3053 (CAN); Flats on W side of Fockler Creek, above spruce forest in creek valley, ca. 2.2 km S of Sandstone Rapids, Coppermine River, 67°25′49″N, 115°37′55″W ± 50 m, 152 m, 1 July 2014, Saarela, Sokoloff & Bull 3121 (CAN, MO); Kugluk (Bloody Falls) Territorial Park, flats above boardwalk W of Bloody Falls, 67°44′34.5″N, 115°22′27″W ± 100 m, 135 m, 13 July 2014, Saarela, Sokoloff & Bull 3923 (CAN, UBC); ca. 0.5 km SW of Heart Lake, SW of Kugluktuk, 7.5 km SW of mouth of Coppermine River, 67°47′52″N, 115°14′14.4″W ± 350 m, 66 m, 23 July 2014, Saarela, Sokoloff & Bull 4284 (ALA, CAN).

Cardamine nymanii Gand., Fig. S21—Cuckoo flower, Meadow bittercress | Circumpolar

Previously recorded from Kugluktuk, as Cardamine pratensis var. angustifolia Hook. (Cody, 1954b; Porsild & Cody, 1980). We made collections at Fockler Creek, Kugluk (Bloody Falls) Territorial Park and Kugluktuk. Widespread throughout the Canadian Arctic (Porsild & Cody, 1980; Korol, 1992; Cody & Reading, 2005; Aiken et al., 2007; Saarela et al., 2013a). Taxonomy follows Al-Shehbaz, Marhold & Lihová (2010), who recognised this taxon as Cardamine nymanii, of which Cardamine pratensis var. angustifolia is a synonym. Elven et al. (2011) provisionally recognised it as Cardamine polemonioides Rouy, a more broadly circumscribed Arctic taxon including Cardamine nymanii.

Specimens Examined: Canada. Nunavut: Kitikmeot Region: Coppermine [Kugluktuk], 67°49′36″N, 115°5′36″W, W. I. Findlay 175 (DAO-184954, not seen); Coppermine [Kugluktuk], between R. C. Mission and DOT [67°49′36″N, 115°5′36″W ± 1.5 km], 12 July 1958, R. D. Wood s.n. (CAN-265540); S of Fockler Creek, along small tributary that runs into Fockler Creek, ca. 2.3 km S of Sandstone Rapids, Coppermine River, 67°25′44.9″N, 115°38′25.9″W ± 100 m, 152 m, 3 July 2014, Saarela, Sokoloff & Bull 3268 (CAN); Sleigh Creek, near its confluence with small, unnamed tributary, ca. 1 km SE of Sandstone Rapids, Coppermine River, 67°27′2″N, 115°37′28.7″W ± 10 m, 150 m, 6 July 2014, Saarela, Sokoloff & Bull 3467 (CAN, MO); Kugluk (Bloody Falls) Territorial Park, rocky valley immediately SW of Bloody Falls, along rough marked section of Portage Trail, head of small unnamed pond just W of falls, 67°44′42.8″N, 115°22′29.2″W ± 10 m, 9 m, 13 July 2014, Saarela, Sokoloff & Bull 3903 (CAN); N side of Heart Lake, below rocky cliff, SW of Kugluktuk, 5.64 km SW of mouth of Coppermine River, 67°48′33.4″N, 115°12′38.8″W ± 25 m, 31 m, 23 July 2014, Saarela, Sokoloff & Bull 4319 (CAN).

Descurainia sophioides (Fisch. ex Hook.) O. E. Schulz—Northern tansy mustard | Asian (N)–amphi-Beringian–North American (NW)

Previously recorded from Kugluktuk (Cody, 1954b; Porsild & Cody, 1980). We made collections of this disturbance-loving species at Sandstone Rapids and Heart Lake. Elsewhere in the Canadian Arctic recorded from Baffin, Banks and Victoria islands, and a few mainland sites (Porsild & Cody, 1980; Korol, 1992; Cody, 1996a; Cody & Reading, 2005; Aiken et al., 2007; Saarela et al., 2013a).

Specimens Examined: Canada. Nunavut: Kitikmeot Region: Coppermine [Kugluktuk], 67°49′36″N, 115°5′36″W, 27 June 1951, W. I. Findlay 39 (ALTA-VP-9646, DAO-184828 01-01000529620, UBC-V40796); Coppermine [Kugluktuk], 67°49′36″N, 115°5′36″W, 4 July 1951, W. I. Findlay 90 (DAO-184829 01-01000529652); Coppermine [Kugluktuk], 67°49′36″N, 115°5′36″W, 8 July 1951, W. I. Findlay 97 (ACAD-30919, DAO-184831 01-01000529651); Coppermine River, Fort Hearne–Bloody Falls [67.7761972°N, 115.2037222°W ± 7.5 km], 1931, A. M. Berry 8 (CAN-63016); Coppermine [Kugluktuk], 67°49′36″N, 115°5′36″W, 27 June 1951, W. I. Findlay 39 (DAO-184828 01-01000529620); Coppermine [Kugluktuk], 67°49′36″N, 115°5′36″W, 4 July 1951, W. I. Findlay 90 (DAO-184829 01-01000529652); Coppermine [Kugluktuk], 67°51′N, 115°16′W, 2 July 1972, F. Fodor N 163 (UBC-V151900); Coppermine [Kugluktuk], near the airstrip [67.8170028°N, 115.1342°W ± 1.2 km], 23 June 1975, F. W. Schueler & J. D. Rising s.n. (CAN-393649); Coppermine [Kugluktuk], old Eskimo [sic] camping places [67°49′36″N, 115°5′36″W ± 1.5 km], 7 July 1958, R. D. Wood s.n. (CAN-265539); Coppermine [Kugluktuk], vic. of hamlet and airstrip, 67.78°N, 115.5°W ± 3,615 m, 23 June 1999, C. L. Parker & I. Jonsdottir 9104 (ALA); Kugluktuk, airport, 21 July 2013, 67.81749°N, 115.13449°W, B. A. Bennett 13-0635 (DAO, det. B. A. Bennett, Dec. 2013); Coppermine River, sandstone cliffs above Sandstone Rapids, 67°27′29.6″N, 115°37′59.3″W ± 100 m, 110 m, 6 July 2014, Saarela, Sokoloff & Bull 3474 (CAN); Heart Lake, SW of Kugluktuk, 6.4 km SW of mouth of Coppermine River, 67°48′6.7″N, 115°13′40.6″W ± 50 m, 41 m, 23 July 2014, Saarela, Sokoloff & Bull 4307 (CAN, MO).

Draba cinerea Adams, Fig. S22—Greyleaf draba | Circumboreal-polar

Previously recorded from Kugluktuk (Cody, 1954b; Porsild & Cody, 1980). We made collections at Fockler Creek, Coppermine Mountains, Kugluk (Bloody Falls) Territorial Park and Kugluktuk and vicinity. Widespread throughout the Canadian Arctic Archipelago and recorded from several other mainland sites (Porsild & Cody, 1980; Cody, Reading & Line, 2003; Aiken et al., 2007; Saarela et al., 2013a).

Specimens Examined: Canada. Nunavut: Kitikmeot Region: Coppermine [Kugluktuk], 67°49′36″N, 115°5′36″W, 15 June 1951, W. I. Findlay 15B (DAO-185055), det. I. A. Al-Shehbaz & S. I. Warwick; second ridge N of Fockler Creek, ca. 1.9 km SSE of Sandstone Rapids, Coppermine River, 67°26′2.4″N, 115°37′26.5″W ± 25 m, 187 m, 2 July 2014, Saarela, Sokoloff & Bull 3215 (CAN, MO); second ridge N of Fockler Creek, ca. 1.9 km SSE of Sandstone Rapids, Coppermine River, 67°26′2.4″N, 115°37′26.5″W ± 25 m, 187 m, 2 July 2014, Saarela, Sokoloff & Bull 3216 (CAN); E side of Fockler Creek, ridge above creek valley before its confluence with Coppermine River, ca. 1.8 km S of Sandstone Rapids, 67°26′3.9″N, 115°38′20.4″W ± 25 m, 168 m, 4 July 2014, Saarela, Sokoloff & Bull 3335 (CAN); flats atop and upper slopes of Coppermine Mountains, N/W side of Coppermine River, 67°14′43.7″N, 115°38′51.2″W ± 150 m, 422 m, 9 July 2014, Saarela, Sokoloff & Bull 3747 (CAN); flats atop and upper slopes of Coppermine Mountains, N/W side of Coppermine River, 67°14′49.9″N, 115°38′43.7″W ± 200 m, 467 m, 9 July 2014, Saarela, Sokoloff & Bull 3771 (CAN); SW-facing slopes of shallow gully in sand hills above Bloody Falls, SE side of Coppermine River across river from Kugluk (Bloody Falls) Territorial Park, 67°44′28.2″N, 115°22′3″W ± 15 m, 78 m, 19 July 2014, Saarela, Sokoloff & Bull 4180 (CAN, UBC); SW-facing slope above Bloody Falls, SE side of Coppermine River, 67°44′27.2″N, 115°22′58″W ± 5 m, 68 m, 19 July 2014, Saarela, Sokoloff & Bull 4199 (CAN); gravel roadside SW of Kugluktuk, S side of road to Heart Lake cemetery, just beyond sewage retention pond, 5.59 km SW of mouth of Coppermine River, 67°48′39″N, 115°12′38.7″W ± 25 m, 46 m, 23 July 2014, Saarela, Sokoloff & Bull 4322 (CAN); gravel roadside SW of Kugluktuk, S side of road to Heart Lake cemetery, just beyond sewage retention pond, 5.59 km SW of mouth of Coppermine River, 67°48′39″N, 115°12′38.7″W ± 25 m, 46 m, 23 July 2014, Saarela, Sokoloff & Bull 4321b (CAN).

Draba fladnizensis Wulfen—Austrian draba | Circumpolar-alpine

Previously recorded from Kugluktuk (Porsild & Cody, 1980). We made collections at Kugluktuk, Coppermine Mountains and Kugluk (Bloody Falls) Territorial Park. It has a scattered distribution in the Canadian Arctic, with records from Baffin, Banks, Cornwallis and Southampton islands, and a few mainland sites (Porsild & Cody, 1980; Korol, 1992; Cody & Reading, 2005; Aiken et al., 2007; Saarela et al., 2013a). A specimen previously determined as this species (Porsild 17178) was re-determined as Draba glabella.

Specimens Examined: Canada. Nunavut: Kitikmeot Region: Coppermine [Kugluktuk], 67°49′36″N, 115°5′36″W, 28 June 1951, W. I. Findlay 44 (DAO-185092 01-01000560616); Kugluktuk, rocky slopes of North Hill, 67°49′31.4″N, 115°6′54″W ± 100 m, 42 m, 29 June 2014, Saarela, Sokoloff & Bull 3072 (CAN); Kugluktuk, rocky slopes of North Hill, 67°49′31.4″N, 115°6′54″W ± 100 m, 42 m, 29 June 2014, Saarela, Sokoloff & Bull 3073 (CAN); E end of small, unnamed lake on W bank of Coppermine River, ca. 8.3 km NNE of Sandstone Rapids, 67°31′30.8″N, 115°36′16.1″W ± 50 m, 126 m, 8 July 2014, Saarela, Sokoloff & Bull 3662 (CAN); flats atop and upper slopes of Coppermine Mountains, N/W side of Coppermine River, 67°14′49.9″N, 115°38′43.7″W ± 200 m, 467 m, 9 July 2014, Saarela, Sokoloff & Bull 3772 (CAN); Kugluk (Bloody Falls) Territorial Park, S-facing cliff (gabbro sill) above start of Bloody Falls, W side of Coppermine River, W side of Portage Trail, 67°44′23.2″N, 115°22′54.5″W ± 50 m, 57 m, 16 July 2014, Saarela, Sokoloff & Bull 4077 (CAN).

Draba glabella Pursh, Fig. S23—Smooth draba | Circumboreal-polar

Previously recorded from Kugluktuk (Cody, 1954b; Porsild & Cody, 1980). We made numerous collections of this common species throughout the study area. It is widespread throughout the Canadian Arctic (Porsild & Cody, 1980; Cody, Scotter & Zoltai, 1989; Korol, 1992; Cody, Reading & Line, 2003; Aiken et al., 2007; Saarela et al., 2013a), but there is a conspicuous gap in the central Canadian Arctic islands; the gap, however, is partially closed by a recent report from Somerset Island (Sokoloff, 2015). Taxonomy follows Al-Shehbaz, Windham & Elven (2010) and Elven et al. (2011). The collection Bennett 13-0646 from near the Kugluktuk Airport was determined by G. A. Mulligan as Draba norvegica Gunnerus, a name of problematic application. The only original material of Draba norvegica is an illustration (the holotype), which corresponds to plants currently recognised as Draba glabella according to Elven et al. (2011), who treated Draba norvegica auct., non Gunnerus as a synonym of the amphi-Atlantic species Draba rupestris W. T. Aiton, the next available name. Porsild & Cody (1980) recognised Draba norvegica and considered Draba rupestris a synonym; they recorded its distribution as primarily amphi-Atlantic, with an outlying dot from the north shore of Great Slave Lake based on two records reported in Mulligan & Cody (1968). Elven et al. (2011) suggested these disjunct records require confirmation. Al-Shehbaz, Windham & Elven (2010) also recognised Draba norvegica in North America, with Draba rupestris as a synonym. Since we have not have examined the Kugluktuk collection, and given the taxonomic uncertainty surrounding application of the name Draba norvegica, we tentatively include this collection here. If confirmed as Draba norvegica (or the taxon currently recognised under that name), it would be the first record for mainland Nunavut and a northwestern range extension.

Specimens Examined: Canada. Nunavut: Kitikmeot Region: Coppermine [Kugluktuk], 67°49′36″N, 115°5′36″W, 27 June 1951, W. I. Findlay 38 (DAO-185120 01-01000560873); Coppermine [Kugluktuk], 67°49′36″N, 115°5′36″W, 15 June 1951, W. I. Findlay 15A (ACAD-30954, ALTA-VP-9296, DAO-185118 01-01000560874); Coppermine [Kugluktuk], vicinity of post [67°49′36″N, 115°5′36″W ± 1.5 km], 26 July 1949, A. E. Porsild 17177 (CAN-127849), 17176 (CAN-127815), 17178 (CAN-127854), 17793 (CAN-127855), 17179 (CAN-127867); Coppermine [Kugluktuk], near the airstrip [67.8170028°N, 115.1342°W ± 1.2 km], 23 June 1975, F. W. Schueler & J. D. Rising s.n. (CAN-393652, CAN-393651); Coppermine [Kugluktuk], near Nursing Station [67°49′36″N, 115°5′36″W ± 1.5 km], 7 July 1958, R. D. Wood s.n. (CAN-265536); Coppermine [Kugluktuk], near old Eskimo [sic] habitation above Anglican Mission [67°49′36″N, 115°5′36″W ± 1.5 km], 7 July 1958, R. D. Wood s.n. (CAN-265533); Kugluktuk, airport, 21 July 2013, 67.81749°N, 115.13449°W, B. A. Bennett 13-0646 (DAO, as Draba norvegica R. Br., det. G. A. Mulligan, April 2014 (see comments above)); Kugluktuk, rocky slopes of North Hill, 67°49′29.6″N, 115°6′31″W ± 50 m, 50 m, 29 June 2014, Saarela, Sokoloff & Bull 3060 (CAN); Kugluktuk, rocky slopes of North Hill, 67°49′31.4″N, 115°6′54″W ± 100 m, 42 m, 29 June 2014, Saarela, Sokoloff & Bull 3074 (CAN, MO); Kugluktuk, disturbed rocky/gravelly ground around house, top of NW-facing gravel slope, just SE of Igalik Building, 67°49′29.6″N, 115°6′31″W ± 10 m, 13 m, 30 June 2014, Saarela, Sokoloff & Bull 3099 (CAN, MO, UBC); old riverbed of Fockler Creek, ca. 2.3 km SSE of Sandstone Rapids, Coppermine River, 67°25′45.7″N, 115°37′21.8″W ± 25 m, 166 m, 2 July 2014, Saarela, Sokoloff & Bull 3176 (ALA, ALTA, CAN); S of Fockler Creek, along small tributary that runs into Fockler Creek, ca. 2.3 km S of Sandstone Rapids, Coppermine River, 67°25′44.9″N, 115°38′25.9″W ± 100 m, 152 m, 3 July 2014, Saarela, Sokoloff & Bull 3248 (CAN); NW-facing slope above tributary of Fockler Creek, just upstream of small tributary from its confluence with Fockler Creek, 67°25′46″N, 115°38′49.4″W ± 200 m, 149 m, 3 July 2014, Saarela, Sokoloff & Bull 3302 (CAN); flats atop and upper slopes of Coppermine Mountains, N/W side of Coppermine River, 67°14′53.6″N, 115°38′37.9″W ± 15 m, 401 m, 9 July 2014, Saarela, Sokoloff & Bull 3756 (CAN, UBC); Kugluk (Bloody Falls) Territorial Park, slope above Bloody Falls (W side) where Coppermine River narrows to Bloody Falls, 67°44′23″N, 115°22′35.9″W ± 25 m, 33 m, 12 July 2014, Saarela, Sokoloff & Bull 3848a (CAN); Kugluk (Bloody Falls) Territorial Park, rocky valley immediately SW of Bloody Falls, along rough marked section of Portage Trail, 67°44′34″N, 115°22′16″W ± 50 m, 20 m, 13 July 2014, Saarela, Sokoloff & Bull 3889 (CAN); Kugluk (Bloody Falls) Territorial Park, sandy NE-facing slope above small creek in deep gully, about 0.5 km W of Bloody Falls, 67°44′36.6″N, 115°22′59.3″W ± 41 m, 41 m, 15 July 2014, Saarela, Sokoloff & Bull 4018 (CAN, MT, O); Kugluk (Bloody Falls) Territorial Park, SE-facing slope above small stream in deep gully that runs into Coppermine River just below Bloody Falls, ca. 1 km W of Bloody Falls, 67°44′41.2″N, 115°23′34.8″W ± 50 m, 49 m, 15 July 2014, Saarela, Sokoloff & Bull 4034 (CAN); W of Kugluktuk on tundra flats above Coppermine River, S of 1 Coronation Drive and N of community power plant, 67°49′28.97″N, 115°5′0.2″W ± 100 m, 8 m, 22 July 2014, Saarela, Sokoloff & Bull 4255 (CAN, US, WIN); gravel roadside SW of Kugluktuk, S side of road to Heart Lake cemetery, just beyond sewage retention pond, 5.59 km SW of mouth of Coppermine River, 67°48′39″N, 115°12′38.7″W ± 25 m, 46 m, 23 July 2014, Saarela, Sokoloff & Bull 4321a (ALA, CAN, MO, UBC); rocky cliffs on S side of Kugluktuk, 67°49′13″N, 115°5′55.8″W ± 50 m, 65 m, 26 July 2014, Saarela, Sokoloff & Bull 4404 (CAN).

Draba lonchocarpa Rydb.—Lance-pod draba | Asian (NE)–amphi-Beringian–Cordilleran | Noteworthy Record

Newly recorded for Nunavut, representing a major range extension for the species. We made collections at Fockler Creek, Kugluk (Bloody Falls) Territorial Park and Heart Lake. They were gathered on dry rocky slopes and outcrops, growing with species such as Anthoxanthum monticola subsp. alpinum, Arnica angustifolia, Betula glandulosa, Calamagrostis purpurascens, Cassiope tetragona, Dasiphora fruticosa, Empetrum nigrum, Poa glauca and Saxifraga tricuspidata. The species was not recorded for Nunavut in Porsild & Cody (1980) and Al-Shehbaz, Windham & Elven (2010). There is a Nunavut specimen (junction of Bailey and Back rivers, 1–2 August 1955, J. S. Tener 293, CAN-235488) originally determined by Porsild as Draba nivalis and re-determined as this species by G. A. Mulligan in 1972. We studied the specimen, and agree with Porsild’s determination. The nearest known site of Draba lonchocarpa is in the Mackenzie Mountains Sekwi Range (Cody, 1978). Taxonomy follows Al-Shehbaz, Windham & Elven (2010). Elven et al. (2011) recognised it as Draba lonchocarpa var. lonchocarpa. One unusual collection (no. 4291) with glabrous leaves and fruits twisted, glabrous and ca. 10 mm is tentatively placed here, on the basis of the twisted fruits; this species typically has pubescent leaf blades.

Specimens Examined: Canada. Nunavut: Kitikmeot Region: E side of Fockler Creek, ridge above creek valley before its confluence with Coppermine River, ca. 1.8 km S of Sandstone Rapids, 67°26′3.9″N, 115°38′20.4″W ± 25 m, 168 m, 4 July 2014, Saarela, Sokoloff & Bull 3336 (CAN); Kugluk (Bloody Falls) Territorial Park, slope above Bloody Falls (W side) where Coppermine River narrows to Bloody Falls, 67°44′23″N, 115°22′35.9″W ± 25 m, 33 m, 12 July 2014, Saarela, Sokoloff & Bull 3848b (CAN); Kugluk (Bloody Falls) Territorial Park, upper ledges of rocky (gabbro) S-facing cliffs above the start of Bloody Falls (W bank of River), just E of Portage Trail, 67°44′21.7″N, 115°22′42.2″W ± 25 m, 46 m, 14 July 2014, Saarela, Sokoloff & Bull 3946 (CAN); Kugluk (Bloody Falls) Territorial Park, SE-facing slope above small stream in deep gully that runs into Coppermine River just below Bloody Falls, ca. 1 km W of Bloody Falls, 67°44′41.2″N, 115°23′34.8″W ± 50 m, 49 m, 15 July 2014, Saarela, Sokoloff & Bull 4035 (CAN); Kugluk (Bloody Falls) Territorial Park, rocky valley immediately SW of Bloody Falls, along rough marked section of Portage Trail, 67°44′34″N, 115°22′16″W ± 50 m, 20 m, 18 July 2014, Saarela, Sokoloff & Bull 4164 (CAN); ca. 0.5 km SW of Heart Lake, SW of Kugluktuk, 7.5 km SW of mouth of Coppermine River, 67°47′52″N, 115°14′14.4″W ± 350 m, 66 m, 23 July 2014, Saarela, Sokoloff & Bull 4291 (CAN).

Draba nivalis Lilj.—Snow draba | Circumpolar-alpine | Noteworthy Record

Newly recorded for the study area. There is a nearby record from the Napaaktoktok River, adjacent to the study area (Cody, 1954b; Porsild & Cody, 1980). Our two collections were gathered in Kugluktuk, where the species grew on dry rock outcrops and ledges. Widely distributed in the Canadian Arctic (Porsild & Cody, 1980; Cody, Scotter & Zoltai, 1989; Korol, 1992; Cody & Reading, 2005; Aiken et al., 2007; Saarela et al., 2013a). Taxonomy follows Al-Shehbaz, Windham & Elven (2010) and Elven et al. (2011).

Specimens Examined: Canada. Nunavut: Kitikmeot Region: Kugluktuk, rocky slopes of North Hill, 67°49′31.4″N, 115°6′54″W ± 100 m, 42 m, 29 June 2014, Saarela, Sokoloff & Bull 3071 (CAN); W of Kugluktuk on tundra flats above Coppermine River, S of 1 Coronation Drive and N of community power plant, 67°49′28.97″N, 115°5′0.2″W ± 100 m, 8 m, 22 July 2014, Saarela, Sokoloff & Bull 4241 (CAN).

Draba pilosa Adams ex DC.—Pilose draba | Asian (N)–amphi-Beringian | Noteworthy Record

This yellow-flowered draba is newly recorded for the study area, even though first collected in Kugluktuk in 1951. That collection was originally identified as Draba alpina L. (Cody, 1954b) and later re-determined as this species (det. G. Mulligan, 2009). We made collections at Fockler Creek, Kugluk (Bloody Falls) Territorial Park and Kugluktuk. It was found in diverse habitats, including tussock tundra, dry tundra along a cliff top, low shrub tundra and a mesic sedge meadow. There are nearby collections made in 1999 from the Big Bend area of the Coppermine River (Reading 2-1Ca, DAO, mixed sheet with Draba cinerea; Reading 10-1, DAO). These collections fill a distribution gap between eastern Great Bear Lake, Tuktut Nogait National Park and vicinity and Bathurst Inlet (Porsild & Cody, 1980; Saarela et al., 2013a). This species is not recorded from the Canadian Arctic Archipelago and is at the northern edge of its range in the study area. Taxonomy follows Al-Shehbaz, Windham & Elven (2010) and Elven et al. (2011).

Specimens Examined: Canada. Nunavut: Kitikmeot Region: Coppermine [Kugluktuk], 67°49′36″N, 115°5′36″W, W. I. Findlay 55 (DAO-185011 01-01000587417); Kugluktuk, disturbed rocky/gravelly ground around house, top of NW-facing gravel slope, just SE of Igalik Building, 67°49′29.6″N, 115°6′31″W ± 10 m, 13 m, 30 June 2014, Saarela, Sokoloff & Bull 3101 (CAN, MO); NW-facing slope just upstream of small tributary from its confluence with Fockler Creek, ca. 2.4 km SSW of Sandstone Rapids, Coppermine River, 67°25′46″N, 115°38′49.4″W ± 200 m, 149 m, 3 July 2014, Saarela, Sokoloff & Bull 3310 (CAN); Coronation Gulf, NW peninsula of Expeditor Cove, ca. 9.6 km NW of Kugluktuk, 67°52′39.1″N, 115°16′43.8″W ± 10 m, 25 m, 8 July 2014, Saarela, Sokoloff & Bull 3701 (ALA, CAN); Kugluk (Bloody Falls) Territorial Park, flats above boardwalk W of Bloody Falls, 67°44′34.5″N, 115°22′27″W ± 100 m, 135 m, 13 July 2014, Saarela, Sokoloff & Bull 3916 (CAN, MO); Kugluk (Bloody Falls) Territorial Park, flats on top of mountain on W side of Coppermine River, just S of the start of Bloody Falls Rapids, 67°44′2.8″N, 115°23′39.3″W ± 250 m, 110 m, 14 July 2014, Saarela, Sokoloff & Bull 3989 (CAN, O); Kugluk (Bloody Falls) Territorial Park, flats on top of mountain on W side of Coppermine River, just S of the start of Bloody Falls Rapids, 67°43′58″N, 115°24′33.3″W ± 25 m, 109 m, 14 July 2014, Saarela, Sokoloff & Bull 4002 (CAN, O).

Draba simmonsii Elven & Al-Shehbaz—Simmons’ draba | North American (N) | Noteworthy Record

Our single collection is the first record for mainland Nunavut and represents a southern range extension in the Central Arctic. At Fockler Creek the species was uncommon and growing along the lower edge of a snowbed community on a north-facing slope with Ranunculus nivalis, Ranunculus pygmaeus, Cardamine bellidifolia and the lichen Masonhalea richardsonii. This yellow-flowered draba was described recently, as part of the Draba micropetala Hook. complex (Elven & Al-Shehbaz, 2008). It was included in a broadly circumscribed Draba alpina in Porsild & Cody (1980). It has its main distribution on the Canadian Arctic islands. Elven & Al-Shehbaz (2008) did not report any collections from mainland Nunavut (there are none at CAN) and they reported one collection from mainland Northwest Territories (Anderson River area, Cape Bathurst), but there are others from these general areas (specimens at CAN).

Specimens Examined: Canada. Nunavut: Kitikmeot Region: flats on W side of Fockler Creek, above spruce forest in creek valley, ca. 2.2 km S of Sandstone Rapids, Coppermine River, 67°25′49″N, 115°37′55″W ± 50 m, 152 m, 1 July 2014, Saarela, Sokoloff & Bull 3138 (CAN).

Erysimum coarctatum Fernald, Fig. 48—Wallflower | North American (N) | Noteworthy Record

Figure 48 Erysimum coarctatum.

(A) Habit, Saarela et al. 3471. (B) Inflorescence, Saarela et al. 4124. Photographs by P. C. Sokoloff (A) and R. D. Bull (B).

Recorded previously from Bloody Falls and Kugluktuk (Cody, 1954b; Porsild & Cody, 1980), as Erysimum inconspicuum (S. Watson) MacMill. and now recognised as this species (Al-Shehbaz, 2010a). Given the revised circumscription, we consider it newly recorded for the study area. We made collections at Sandstone Rapids, Kugluk (Bloody Falls) Territorial Park and Heart Lake. It grows in disturbed areas, including on slopes and sandstone cliffs and along ridges and creek beds. At some sites, populations comprised only a few scattered plants while at others the species was locally common. Keys to distinguish Erysimum coarctatum and Erysimum inconspicuum are given in Al-Shehbaz (2010a) and Saarela et al. (2013a). On the mainland Canadian Arctic known from the Lower Brock River area (Saarela et al., 2013a) and as far east as Bathurst Inlet (Porsild & Cody, 1980) and Hood River (Gould & Walker, 1997). It is not recorded for the Canadian Arctic Archipelago and not reported for Nunavut in Al-Shehbaz (2010a), even though previously documented there.

Specimens Examined: Canada. Nunavut: Kitikmeot Region: Bloody Falls on Coppermine River, 67°44′N, 115°23′W, 27 July 1951, W. I. Findlay 203 (DAO-185399 01-01000581796); Coppermine [Kugluktuk], 67°49′36″N, 115°5′36″W, 4 July 1951, W. I. Findlay 87 (DAO-185398 01-01000581797); Coppermine River, east bank, 10 July 1955, R. E. Miller 87 (CAN-241981); second ridge N of Fockler Creek, ca. 1.9 km SSE of Sandstone Rapids, Coppermine River, 67°26′2.4″N, 115°37′26.5″W ± 25 m, 187 m, 2 July 2014, Saarela, Sokoloff & Bull 3210 (CAN); N side of Fockler Creek, ca. 1.9 km S of Sandstone Rapids, Coppermine River, 67°25′57.89″N, 115°38′3.9″W ± 10 m, 162 m, 4 July 2014, Saarela, Sokoloff & Bull 3325 (CAN); S side of Fockler Creek, ca. 2.7 SE of Sandstone Rapids, Coppermine River, 67°25′38.2″N, 115°36′54.9″W ± 50 m, 128 m, 5 July 2014, Saarela, Sokoloff & Bull 3402 (CAN, UBC); Coppermine River, sandstone cliffs above Sandstone Rapids, 67°27′29.6″N, 115°37′59.3″W ± 100 m, 110 m, 6 July 2014, Saarela, Sokoloff & Bull 3471 (ALTA, CAN, MO); Kugluk (Bloody Falls) Territorial Park, S-facing cliff (gabbro sill) above start of Bloody Falls, W side of Coppermine River, W side of Portage Trail, 67°44′23.2″N, 115°22′54.5″W ± 50 m, 57 m, 16 July 2014, Saarela, Sokoloff & Bull 4085 (CAN); Kugluk (Bloody Falls) Territorial Park, rocky sandy beach just below Bloody Falls, W side of Coppermine River, vicinity of confluence with small creek, beach seasonally flooded, 67°44′54.5″N, 115°22′17.2″W ± 75 m, 9 m, 17 July 2014, Saarela, Sokoloff & Bull 4124 (ALA, CAN); gravel roadside SW of Kugluktuk, S side of road to Heart Lake cemetery, just beyond sewage retention pond, 5.59 km SW of mouth of Coppermine River, 67°48′39″N, 115°12′38.7″W ± 25 m, 46 m, 23 July 2014, Saarela, Sokoloff & Bull 4324 (CAN).

Erysimum pallasii (Pursh) Fernald—Pallas’ wallflower | Asian (N)–amphi-Beringian–North American (N) | Noteworthy Record

Our single collection is the first record for the study area, closing a distribution gap on the mainland between Great Slave Lake and Bathurst Inlet (Porsild & Cody, 1980). We encountered this conspicuous pink-flowered species growing on manufactured gravel slopes around Kugluktuk’s sewage retention pond, growing with Artemisia tilesii, Chamerion latifolium, Descurainia sophioides, Erysimum coarctatum, Poa glauca and Puccinellia nuttalliana. A nearby collection was made on the shore of an island in Coronation Gulf northeast of the mouth of the Coppermine River (Findlay 101, DAO; Cody, 1954b; Porsild & Cody, 1980). Elsewhere in the Canadian Arctic known from northern Baffin Island, Banks, Ellesmere, Melville and Victoria islands, and western mainland sites as far east as the Bathurst Inlet area (Porsild & Cody, 1980; Cody & Reading, 2005; Aiken et al., 2007; Saarela et al., 2013a); there is also a record from southeastern mainland Nunavut (Cody & Reading, 2005).

Specimens Examined: Canada. Nunavut: Kitikmeot Region: manufactured gravel slopes around Kugluktuk’s sewage retention pond, 5.16 km SW of Coppermine River, 67°48′52.38″N, 115°12′10.3″W ± 10 m, 35 m, 23 July 2014, Saarela, Sokoloff & Bull 4330 (CAN, UBC).

Eutrema edwardsii R. Br.—Edward’s eutrema | Circumpolar–alpine | Noteworthy Record

Newly recorded for the study area. We made two collections in the Fockler Creek area, which close a distribution gap between Bernard Harbour and a site to the east of the study area (Porsild & Cody, 1980). Widespread in the Canadian Arctic (Porsild & Cody, 1980; Cody, Scotter & Zoltai, 1989; Korol, 1992; Cody & Reading, 2005; Aiken et al., 2007; Saarela et al., 2013a).

Specimens Examined: Canada. Nunavut: Kitikmeot Region: just above second ridge N of Fockler Creek, ca. 1.9 km SSE of Sandstone Rapids, Coppermine River, 67°26′3.3″N, 115°37′25.6″W ± 5 m, 169 m, 2 July 2014, Saarela, Sokoloff & Bull 3219 (CAN); NW-facing slope just upstream of small tributary from its confluence with Fockler Creek, ca. 2.4 km SSW of Sandstone Rapids, Coppermine River, 67°25′46″N, 115°38′49.4″W ± 200 m, 149 m, 3 July 2014, Saarela, Sokoloff & Bull 3303 (CAN).

Physaria arctica (Wormsk. ex Hornem.) O’Kane & Al-Shehbaz, Fig. 49—Arctic bladderpod | Asian (N)–amphi-Beringian–North American (N) | Noteworthy Record

Figure 49 Physaria arctica.

(A) Habit, Saarela et al. 4177. (B) Fruits, Saarela et al. 4177. Photographs by P. C. Sokoloff.

Newly recorded for the study area. We made collections at Big Creek, Muskox Rapids and Bloody Falls. On mainland Nunavut this circumpolar Arctic-alpine taxon is known only from a few records on the Adelaide and Boothia Peninsulas (Rollins & Shaw, 1973; Porsild & Cody, 1980, as Lesquerella arctica Wormsk. ex Hornem.). In the western Arctic, the nearest mainland collections are from eastern Great Bear Lake and adjacent to Dolphin and Union Straight in western Nunavut (Rollins & Shaw, 1973; Porsild & Cody, 1980). In the Canadian Arctic Archipelago known from Baffin, Banks, Coats, Devon, Ellesmere, Southampton and Victoria islands (Aiken et al., 2007).

Specimens Examined: Canada. Nunavut: Kitikmeot Region: forest and slopes at confluence of Big Creek and Coppermine River, N side of Coppermine River, S side of Coppermine Mountains, 67°14′29.3″N, 116°2′44.5″W ± 250 m, 180–199 m, 7 July 2014, Saarela, Sokoloff & Bull 3551 (CAN, MO); esker on E side of Coppermine River, 0.6 km SSE of Muskox Rapids, 67°22′40″N, 115°42′38.5″W ± 50 m, 172 m, 7 July 2014, Saarela, Sokoloff & Bull 3610 (CAN); top of hill on E side of Bloody Falls, across Coppermine River from Kugluk (Bloody Falls) Territorial Park, 67°44′28.2″N, 115°22′0.3″W ± 50 m, 78 m, 19 July 2014, Saarela, Sokoloff & Bull 4177 (ALA, ALTA, CAN, MO, O).

Transberingia bursifolia (DC.) Al-Shehbaz & O’Kane subsp. bursifolia—Soft fissurewort | Asian (NE)–amphi-Beringian–North American (N)

Previously recorded from Kugluktuk, where found growing along and at the base of a cliff face (Cody, 1954b; Porsild & Cody, 1980). We did not encounter this species in 2014. There are few records from elsewhere in the Canadian Arctic, from Banks, Baffin and Ellesmere islands (Aiken et al., 2007) and it was recently reported from Cambridge Bay, the first record for Victoria Island (Bennett, 2015). There are no mainland records east of the study area. It was previously treated as Halimolobos mollis (Hook.) Rollins, but is now recognised in Transberingia Al-Shehbaz & O’Kane (Al-Shehbaz & O’Kane, 2003; Al-Shehbaz, 2010b; Elven et al., 2011), with two subspecies. Only the nominate subspecies occurs in the Arctic (Al-Shehbaz, 2010b).

Specimens Examined: Canada. Nunavut: Kitikmeot Region: Coppermine [Kugluktuk], 67°49′36″N, 115°5′36″W, 23 June 1951, W. I. Findlay 33 (DAO-185260 01-01000576171); Coppermine [Kugluktuk], 67°49′36″N, 115°5′36″W, 7 June 1951, W. I. Findlay 7 (DAO-185263 01-01000576172).

Campanulaceae [1/1]

Campanula uniflora L.—Arctic harebell | Amphi-Beringian–North American (N)–amphi-Atlantic | Noteworthy Record

Newly recorded for the study area. We made collections at Fockler Creek and Kugluk (Bloody Falls) Territorial Park in low shrub tundra, on dry rocky slopes, on dry windswept tundra and in a mesic sedge meadow. These close a small gap between Bernard Harbour and Epworth Harbour (Macoun & Holm, 1921; Porsild & Cody, 1980). Elsewhere in the Canadian Arctic known from Baffin, Banks, Coats, Devon, Ellesmere, Melville, Somerset, Southampton and Victoria islands, and mainland sites (Porsild & Cody, 1980; Cody, Scotter & Zoltai, 1989; Korol, 1992; Cody & Reading, 2005; Aiken et al., 2007; Saarela et al., 2013a).

Specimens Examined: Canada. Nunavut: Kitikmeot Region: N side of Fockler Creek, ca. 1.9 km S of Sandstone Rapids, Coppermine River, 67°25′57.89″N, 115°38′3.9″W ± 10 m, 162 m, 4 July 2014, Saarela, Sokoloff & Bull 3324 (CAN); SW-facing slope above (N side) of Fockler Creek, ca. 3.2 km SE of Sandstone Rapids, Coppermine River, 67°25′26.2″N, 115°36′14″W ± 25 m, 193 m, 5 July 2014, Saarela, Sokoloff & Bull 3412 (CAN); ridge top N of Fockler Creek and S of Tundra Lake, ca. 3.8 km SE of Sandstone Rapids, Coppermine River, 67°25′20.4″N, 115°34′17.2″W ± 3 m, 273 m, 5 July 2014, Saarela, Sokoloff & Bull 3424 (CAN); Kugluk (Bloody Falls) Territorial Park, flats on top of mountain on W side of Coppermine River, just S of the start of Bloody Falls Rapids, 67°44′2.8″N, 115°23′39.3″W ± 250 m, 110 m, 14 July 2014, Saarela, Sokoloff & Bull 3992 (CAN).

Caryophyllaceae [9/18]

Arenaria humifusa Wahlenb., Fig. 50A—Creeping sandwort | North American (N)–amphi-Atlantic

Figure 50 Arenaria humifusa, Eremogone capillaris subsp. capillaris and Honckenya peploides subsp. diffusa.

Arenaria humifusa: (A) habit, Saarela et al. 3264. Eremogone capillaris subsp. capillaris: (B) habit, Saarela et al. 3570. Honckenya peploides subsp. diffusa: (C) habit, Saarela et al. 3708. Photographs by P. C. Sokoloff.

Previously recorded from Kugluktuk (Porsild & Cody, 1980). We made collections at Fockler Creek, Sleigh Creek and Big Creek. Elsewhere in the Canadian Arctic recorded from Baffin, Coats, Cornwallis, Ellesmere, Southampton and Victoria islands, mainland sites in Northwest Territories and Nunavut, and northern Quebec and Labrador (Porsild & Cody, 1980; Korol, 1992; Aiken et al., 2007; Saarela et al., 2013a; L. J. Gillespie & J. M. Saarela, 2016, unpublished data; Blondeau, 2015a).

Specimens Examined: Canada. Nunavut: Kitikmeot Region: Coppermine [Kugluktuk], vicinity of post [67°49′36″N, 115°5′36″W ± 1.5 km], 26 July 1949, A. E. Porsild 17171 (CAN-127678); S of Fockler Creek, along small tributary that runs into Fockler Creek, ca. 2.3 km S of Sandstone Rapids, Coppermine River, 67°25′44.9″N, 115°38′25.9″W ± 100 m, 152 m, 3 July 2014, Saarela, Sokoloff & Bull 3264 (CAN); Sleigh Creek, near its confluence with small, unnamed tributary, ca. 1 km SE of Sandstone Rapids, Coppermine River, 67°27′2″N, 115°37′28.7″W ± 10 m, 150 m, 6 July 2014, Saarela, Sokoloff & Bull 3468 (CAN); forest and slopes at confluence of Big Creek and Coppermine River, N side of Coppermine River, S side of Coppermine Mountains, 67°14′29.3″N, 116°2′44.5″W ± 250 m, 180–199 m, 7 July 2014, Saarela, Sokoloff & Bull 3544 (CAN).

Cerastium alpinum L. subsp. alpinum—Alpine chickweed | Amphi-Atlantic (W)

Previously recorded from Kugluktuk by Cody (1954b), but not mapped in Porsild & Cody (1980). We made collections at Fockler Creek, Melville Creek and Kugluk (Bloody Falls) Territorial Park. The Cerastium alpinum–Cerastium arcticum Lange complex is a taxonomically complicated polyploid group (Brysting & Elven, 2000; Elven et al., 2011). Taxonomy here follows Morton (2005a). In Elven et al. (2011) our plants correspond to Cerastium alpinum “arctic race”, an unnamed taxon they provisionally accept. The global range given is for the species, not the infraspecific taxon, since there are differing circumscriptions of infraspecific taxa in recent literature. Aiken et al. (2007) record it from eastern islands in the Canadian Arctic Archipelago and a few mainland Nunavut sites, and it is widespread in northern Quebec and Labrador (Blondeau, 2015a).

Specimens Examined: Canada. Nunavut: Kitikmeot Region: Coppermine [Kugluktuk], 67°49′36″N, 115°5′36″W, 20 June 1951, W. I. Findlay 16 (DAO-183644 01-01000617018); old riverbed of Fockler Creek, ca. 2.3 km SSE of Sandstone Rapids, Coppermine River, 67°25′45.7″N, 115°37′21.8″W ± 25 m, 166 m, 2 July 2014, Saarela, Sokoloff & Bull 3179 (CAN); confluence of Coppermine River and Melville Creek, just W of Coppermine Mountains, 67°15′52″N, 115°30′55.3″W ± 350 m, 178–190 m, 7 July 2014, Saarela, Sokoloff & Bull 3497 (ALA, ALTA, CAN, UBC); Kugluk (Bloody Falls) Territorial Park, rocky sandy beach just below Bloody Falls, W side of Coppermine River, vicinity of confluence with small creek, beach seasonally flooded, 67°44′54.5″N, 115°22′17.2″W ± 75 m, 9 m, 17 July 2014, Saarela, Sokoloff & Bull 4120 (CAN).

Cerastium beeringianum Cham. & Schltdl.—Bering Sea chickweed | Asian (N)–amphi-Beringian–North American (N)

Previously recorded from Bloody Falls and Kugluktuk (Cody, 1954b; Porsild & Cody, 1980). We made collections at Fockler Creek, Kugluk (Bloody Falls) Territorial Park and on an island in the mouth of the Coppermine River. Taxonomy follows Morton (2005a), who does not accept infraspecific taxa. Our plants likely correspond to Cerastium beeringianum var. beeringianum in Elven et al. (2011). This is the common chickweed species in the area. Elsewhere in the Canadian Arctic widespread in western Arctic islands, recorded from southern sites in the eastern Arctic islands, and recorded from numerous mainland sites (Porsild & Cody, 1980; Cody & Reading, 2005; Aiken et al., 2007; Saarela et al., 2013a; Blondeau, 2015a).

Specimens Examined: Canada. Nunavut: Kitikmeot Region: Bloody Falls on Coppermine River, 67°44′N, 115°23′W, 27 July 1951, W. I. Findlay 204 (DAO-183660 01-01000617031); Coppermine [Kugluktuk], 67°49′36″N, 115°5′36″W, 16 July 1951, W. I. Findlay 130 (DAO-183643 01-01000617032) (Porsild & Cody, 1980; Cody & Reading, 2005; Aiken et al., 2007; Saarela et al., 2013a; Blondeau, 2015a), 4 July 1951, W. I. Findlay 91 (DAO-183642 01-01000617028) (Porsild & Cody, 1980; Cody & Reading, 2005; Aiken et al., 2007; Saarela et al., 2013a; Blondeau, 2015a), 2 July 1951, W. I. Findlay 63 (DAO-183657 01-01000617033, UBC-V40797); Kugluktuk, disturbed rocky/gravelly ground around house, top of NW-facing gravel slope, just SE of Igalik Building, 67°49′29.6″N, 115°6′31″W ± 10 m, 13 m, 30 June 2014, Saarela, Sokoloff & Bull 3098 (CAN); E side of Fockler Creek, in valley just above creek’s confluence with the Coppermine River, ca. 1.4 km SSW of Sandstone Rapids, 67°26′21.4″N, 115°38′54″W ± 5 m, 140 m, 4 July 2014, Saarela, Sokoloff & Bull 3360 (CAN, UBC); slopes on E side of Coppermine River, N of its confluence with Fockler Creek, ca. 0.8 km SW of Sandstone Rapids, 67°26′36.9″N, 115°38′50.1″W ± 50 m, 128 m, 4 July 2014, Saarela, Sokoloff & Bull 3380 (CAN, MO, MT, O); S-facing sandstone cliffs above Coppermine River, ca. 7.8 km NNE of Sandstone Rapids, 67°31′15.1″N, 115°36′19.1″W ± 50 m, 106 m, 8 July 2014, Saarela, Sokoloff & Bull 3638 (CAN, K, NY, QFA, US, WIN); unnamed island just E (ca. 3.3 km) of Kugluktuk at mouth of Coppermine River, 67°49′29.2″N, 115°1′3.2″W ± 50 m, 1 m, 8 July 2014, Saarela, Sokoloff & Bull 3717 (ALA, ALTA, CAN); Kugluk (Bloody Falls) Territorial Park, along wet, muddy, and deeply pitted ATV trail ca. 1 km W of Bloody Falls, 67°44′33.2″N, 115°23′30″W ± 20 m, 73 m, 16 July 2014, Saarela, Sokoloff & Bull 4099 (CAN).

Eremogone capillaris (Poir.) Fenzl subsp. capillaris, Fig. 50B—Thread-leaved sandwort | Asian (C-NE)–amphi-Beringian | Noteworthy Record

First record for the study area and Nunavut, and a northeastern range extension from the nearest collections from Great Bear Lake (Porsild, 1943, as Arenaria nardifolia Ledeb.; Porsild & Cody, 1980, as Arenaria capillaris var. nardifolia (Ledeb.) Reg.). We found this species on a dry, sandy, gravelly and sparsely vegetated terrace on the south side of the Kendall River growing with Arctous rubra. We did not find it in the Arctic portion of the study area, but it reaches the Canadian Arctic in the Mackenzie Delta area (Porsild & Cody, 1980). Arenaria subg. Eremogone (Fenzl) Fenzl, which includes this species, is now recognised as the genus Eremogone Fenzl, a circumscription supported by molecular evidence (Harbaugh et al., 2010).

Specimens Examined: Canada. Nunavut: Kitikmeot Region: confluence of Coppermine and Kendall rivers (NW side of Coppermine River, S side of Kendall River), 67°6′51.1″N, 116°8′18.3″W ± 150 m, 220 m, 7 July 2014, Saarela, Sokoloff & Bull 3570 (ALA, ALTA, CAN, O, UBC).

Honckenya peploides subsp. diffusa (Hornem.) Hultén ex V. V. Petrovsky, Fig. 50C—Beach sandwort, seabeach sandwort, beach greens | Circumpolar

Previously recorded from Kugluktuk (Cody, 1954b, as Arenaria peploides L.; Porsild & Cody, 1980). We made collections of this seashore species on an island at the mouth of the Coppermine River and along the coast west of Kugluktuk. Elsewhere in the Canadian Arctic recorded from Baffin, Banks, Coats, King William, Salisbury, Southampton and Victoria islands, and numerous mainland sites (Porsild & Cody, 1980; Cody, Scotter & Zoltai, 1989; Korol, 1992; Aiken et al., 2007; Saarela et al., 2013a; Blondeau, 2015a). Taxonomy follows Elven et al. (2011).

Specimens Examined: Canada. Nunavut: Kitikmeot Region: Coppermine [Kugluktuk], 67°49′36″N, 115°5′36″W, 13 July 1951, W. I. Findlay 115 (DAO-183772 01-01000616641) (Elven et al., 2011), W. I. Findlay 59 (ACAD-30918, DAO-183773 01-01000616642, UBC-V40798); Coppermine [Kugluktuk], 67°49′36″N, 115°5′36″W, 2 August 1995, T. Dolman 90 (LEA); unnamed island just E (ca. 3.3 km) of Kugluktuk at mouth of Coppermine River, 67°49′29.2″N, 115°1′3.2″W ± 50 m, 1 m, 8 July 2014, Saarela, Sokoloff & Bull 3708 (ALA, ALTA, CAN); sandy beach along Coronation Gulf, 3.9 km W of mouth of Coppermine River, 67°49′37.8″N, 115°10′31.8″W ± 50 m, 9 m, 23 July 2014, Saarela, Sokoloff & Bull 4337 (CAN, UBC).

Minuartia biflora (L.) Schinz & Thell.—Mountain stitchwort | Circumpolar–alpine

Previously recorded from Kugluktuk (Porsild & Cody, 1980), but we were unable to locate a voucher for confirmation. We made collections at Fockler Creek, Escape Rapids and Coppermine Mountains. Elsewhere in the Canadian Arctic recorded from Baffin, Banks, Ellesmere, Somerset and Victoria islands, some mainland sites in Nunavut and Northwest Territories, and numerous sites in northern Quebec and Labrador (Porsild & Cody, 1980; Cody, Scotter & Zoltai, 1984; Cody & Reading, 2005; Aiken et al., 2007; Saarela et al., 2013a; Blondeau, 2015a). The recent transfer of species formerly treated in the polyphyletic Minuartia L. to the genera Cherleria, Pseudocherleria Dillenb. & Kadereit, Mononeuria Rchb. and Sabulina Rchb. (Dillenberger & Kadereit, 2014) in the context of the Canadian Arctic flora is reviewed in Gillespie et al. (2015). Minuartia biflora is part of a clade of Caryophyllaceae now recognised as the genus Cherleria L., but a combination in that genus is not yet available.

Specimens Examined: Canada. Nunavut: Kitikmeot Region: old riverbed of Fockler Creek, ca. 2.3 km SSE of Sandstone Rapids, Coppermine River, 67°25′45.7″N, 115°37′21.8″W ± 25 m, 166 m, 1 July 2014, Saarela, Sokoloff & Bull 3156 (CAN); NW-facing slope above tributary of Fockler Creek, ca. 2.4 km SSW of Sandstone Rapids, Coppermine River, 67°25′46″N, 115°38′49.4″W ± 50 m, 149 m, 3 July 2014, Saarela, Sokoloff & Bull 3300b (CAN); SE-facing slopes above Escape Rapids, W side of Coppermine River, 67°36′58.7″N, 115°29′18.3″W ± 99 m, 50 m, 8 July 2014, Saarela, Sokoloff & Bull 3732 (CAN); flats atop and upper slopes of Coppermine Mountains, N/W side of Coppermine River, 67°14′53.6″N, 115°38′37.9″W ± 15 m, 401 m, 9 July 2014, Saarela, Sokoloff & Bull 3759 (CAN); flats atop and upper slopes of Coppermine Mountains, N/W side of Coppermine River, 67°14′49.9″N, 115°38′43.7″W ± 200 m, 467 m, 9 July 2014, Saarela, Sokoloff & Bull 3774 (CAN).

Sabulina elegans (Cham. & Schltdl.) Dillenb. & Kadereit, Fig. 51—Elegant stitchwort | Amphi-Beringian–Cordilleran | Noteworthy Record

Figure 51 Sabulina elegans.

(A) Habit, Saarela et al. 4328. (B) Inflorescence, Saarela et al. 4328. Photographs by P. C. Sokoloff.

First report of the species for the study area and Nunavut. We made collections at Fockler Creek, Big Creek and west of Kugluktuk. The species was uncommon at each of our three collection sites. One collection from Kugluktuk (Findlay 252) and two from the nearby Big Bend area of the Coppermine River (Reading 3-1, 31-1, DAO) (Cody, Reading & Line, 2003) originally determined as Minuartia rossii have been re-determined as this species (det. J. M. Saarela, 2015). All of these collections represent an eastern range extension from the nearest known sites in Tuktut Nogait National Park and vicinity (Saarela et al., 2013a). The Coppermine River area is now the known eastern limit of this species. Elsewhere in the Canadian Arctic known from mainland Northwest Territories and northwestern Victoria Island (Aiken et al., 2007; L. J. Gillespie & J. M. Saarela, 2016, unpublished data).

Species treated in Sabulina were recently transferred from the polyphyletic genus Minuartia L. (Dillenberger & Kadereit, 2014), as reviewed in Gillespie et al. (2015). Species circumscriptions follow authorities (Wolf, Packer & Denford, 1979; Rabeler, Hartman & Utech, 2005; Elven et al., 2011) who recognise Sabulina elegans (syn. Minuartia elegans (Cham. & Schltdl.) Schischk.) as distinct from Sabulina rossii (syn. Minuartia rossii (R. Br. ex Richardson) Graebn.), and a key distinguishing them is given in Saarela et al. (2013a). Porsild & Cody (1980) treated this species as Minuartia rossii subsp. elegans (Cham. & Schltdl.) Rebr. Wolf, Packer & Denford (1979) considered Sabulina elegans to occur only in Alaska and Yukon, and treated all Northwest Territories and Nunavut specimens as Sabulina rossii. However, most collections of the complex from Tuktut Nogait National Park and vicinity in eastern Northwest Territories were recently found to be Sabulina elegans (Saarela et al., 2013a), and we also made three collections in the study area. In Canada, Sabulina rossii appears to be a more northern species with a main distribution in the Arctic islands and a few mainland Northwest Territories and Nunavut localities, while Sabulina elegans is a more southern species with a main distribution in Alaska and Yukon, extending to the study area.

Specimens Examined: Canada. Nunavut: Kitikmeot Region: Coppermine [Kugluktuk], 67°49′36″N, 115°5′36″W, 4 August 1951, W. I. Findlay 252 (DAO-1725 01-01000620116); S of Fockler Creek, above small tributary of Fockler Creek, ca. 2.3 km S of Sandstone Rapids, Coppermine River, 67°25′46.3″N, 115°38′2.5″W ± 100 m, 156 m, 6 July 2014, Saarela, Sokoloff & Bull 3455 (CAN); forest and slopes at confluence of Big Creek and Coppermine River, N side of Coppermine River, S side of Coppermine Mountains, 67°14′29.3″N, 116°2′44.5″W ± 250 m, 180–199 m, 7 July 2014, Saarela, Sokoloff & Bull 3563 (CAN); manufactured gravel slopes around Kugluktuk’s sewage retention pond, 5.16 km SW of Coppermine River, 67°48′52.38″N, 115°12′10.3″W ± 10 m, 35 m, 23 July 2014, Saarela, Sokoloff & Bull 4328 (CAN).

Sabulina rossii (R. Br.) Dillenb. & Kadereit—Ross’s sandwort | Amphi-Beringian (E)–North American (N)–amphi-Atlantic (W) | Noteworthy Record

Our single collection from Fockler Creek is the first confirmed record for the study area and for western mainland Nunavut, according to the map in Aiken et al. (2007), and a southern range extension in the Central Arctic. The collection was gathered on dry, hummocky mesic tundra dominated by frost boils and associated with Betula glandulosa, Hedysarum alpinum, Lupinus arcticus, Picea glauca, Pinguicula vulgaris, Rhododendron lapponicum, Salix spp. and Vaccinium uliginosum. Previous collections from the study area reported as this species have been re-determined as Sabulina elegans and Sabulina stricta (see comments under those species). Taxonomy is reviewed under Sabulina elegans. This is primarily a high Arctic species distributed throughout most of the Canadian Arctic Archipelago and known from a few, mostly northern, mainland sites in Nunavut and Northwest Territories, and a single site in northern Quebec (Porsild & Cody, 1980; Cody & Reading, 2005; Aiken et al., 2007; Saarela et al., 2013a; Blondeau, 2015a).

Specimens Examined: Canada. Nunavut: Kitikmeot Region: E side of Fockler Creek, in valley just above creek’s confluence with the Coppermine River, ca. 1.4 km SSW of Sandstone Rapids, 67°26′14.5″N, 115°38′34.8″W ± 50 m, 146 m, 4 July 2014, Saarela, Sokoloff & Bull 3349 (CAN, UBC).

Sabulina rubella (Wahlenb.) Dillenb. & Kadereit, Fig. S24—Beautiful sandwort, reddish sandwort | Circumpolar–alpine

Previously recorded from Kugluktuk (Cody, 1954b; Porsild & Cody, 1980). We made numerous collections at Fockler Creek, Bigtree River, south of Escape Rapids, Kugluk (Bloody Falls) Territorial Park, Heart Lake and Kugluktuk. This is the most common species of Sabulina in the study area. It was previously recognised as Minuartia rubella Wahlenb. (Porsild & Cody, 1980; Rabeler, Hartman & Utech, 2005; Elven et al., 2011). Widespread throughout the Canadian Arctic (Porsild & Cody, 1980; Cody, Scotter & Zoltai, 1989; Korol, 1992; Cody & Reading, 2005; Aiken et al., 2007; Saarela et al., 2013a; Blondeau, 2015a).

Specimens Examined: Canada. Nunavut: Kitikmeot Region: Coppermine [Kugluktuk], vicinity of post [67°49′36″N, 115°5′36″W ± 1.5 km], 26 July 1949, A. E. Porsild 17172 (CAN-127695); Kugluktuk, disturbed rocky/gravelly ground around house, top of NW-facing gravel slope, just SE of Igalik Building, 67°49′29.6″N, 115°6′31″W ± 10 m, 13 m, 30 June 2014, Saarela, Sokoloff & Bull 3096 (CAN, MO, UBC); old riverbed of Fockler Creek, ca. 2.3 km SSE of Sandstone Rapids, Coppermine River, 67°25′45.7″N, 115°37′21.8″W ± 25 m, 166 m, 2 July 2014, Saarela, Sokoloff & Bull 3172 (CAN); second ridge N of Fockler Creek, ca. 1.9 km SSE of Sandstone Rapids, Coppermine River, 67°26′2.4″N, 115°37′26.5″W ± 25 m, 187 m, 2 July 2014, Saarela, Sokoloff & Bull 3212 (CAN); E side of Fockler Creek, ridge above creek valley before its confluence with Coppermine River, ca. 1.8 km S of Sandstone Rapids, 67°26′3.9″N, 115°38′20.4″W ± 25 m, 168 m, 4 July 2014, Saarela, Sokoloff & Bull 3338 (CAN); slopes on E side of Coppermine River, N of its confluence with Fockler Creek, ca. 0.8 km SW of Sandstone Rapids, 67°26′36.9″N, 115°38′50.1″W ± 50 m, 128 m, 4 July 2014, Saarela, Sokoloff & Bull 3395 (CAN); SW-facing slope above (N side) of Fockler Creek, ca. 3.2 km SE of Sandstone Rapids, Coppermine River, 67°25′26.2″N, 115°36′14″W ± 25 m, 193 m, 5 July 2014, Saarela, Sokoloff & Bull 3411 (ALA, CAN); confluence of Coppermine and Bigtree rivers, 66°56′23.8″N, 116°21′3.2″W ± 100 m, 265 m, 7 July 2014, Saarela, Sokoloff & Bull 3606 (CAN); S-facing slopes on W side of Coppermine River, about halfway between Escape Rapids and Muskox Rapids, 67°31′18.2″N, 115°36′20.1″W ± 150 m, 115 m, 8 July 2014, Saarela, Sokoloff & Bull 3625 (CAN); Kugluk (Bloody Falls) Territorial Park, slope above Bloody Falls (W side) where Coppermine River narrows to Bloody Falls, 67°44′23″N, 115°22′35.9″W ± 25 m, 33 m, 12 July 2014, Saarela, Sokoloff & Bull 3847 (CAN); Kugluk (Bloody Falls) Territorial Park, upper ledges of rocky (gabbro) S-facing cliffs above the start of Bloody Falls (W bank of River), just E of Portage Trail, 67°44′21.7″N, 115°22′42.2″W ± 25 m, 46 m, 16 July 2014, Saarela, Sokoloff & Bull 4069 (CAN); Kugluk (Bloody Falls) Territorial Park, W side of Coppermine River, along ATV trail below slope of sand hill just below picnic bench/lookout area, 67°44′41.5″N, 115°22′14.9″W ± 15 m, 15 m, 17 July 2014, Saarela, Sokoloff & Bull 4151 (CAN); Heart Lake, SW of Kugluktuk, 6.4 km SW of mouth of Coppermine River, 67°48′6.7″N, 115°13′40.6″W ± 50 m, 41 m, 23 July 2014, Saarela, Sokoloff & Bull 4308 (CAN); manufactured gravel slopes around Kugluktuk’s sewage retention pond, 5.16 km SW of Coppermine River, 67°48′52.38″N, 115°12′10.3″W ± 10 m, 35 m, 23 July 2014, Saarela, Sokoloff & Bull 4329 (CAN); SE edge of Kugluktuk, rocky cliffs overlooking Coppermine River, 67°49′9.2″N, 115°5′40.4″W ± 50 m, 28 m, 24 July 2014, Saarela, Sokoloff & Bull 4357 (ALTA, CAN).

Sabulina stricta (Sw.) Rchb.—Bog sandwort | Circumpolar–alpine | Noteworthy Record

Newly recorded for the study area and western mainland Nunavut. We made collections at Fockler Creek and Melville Creek. It was found in wet disturbed ground in mesic shrub tundra, a snowbed habitat, on a rocky creek shore and in a meadow around a small pond. Elsewhere in Nunavut known from the southeastern mainland and Baffin, Southampton and Victoria islands (Porsild & Cody, 1980; Aiken et al., 2007; Gillespie et al., 2015), and there are numerous collections from northern Quebec (Blondeau, 2015a). There is a conspicuous distribution gap in the central Canadian Arctic. The closest collections to the west are from Tuktut Nogait National Park and vicinity (Saarela et al., 2013a). The species is at the northern edge of its range in the study area. It was previously recognised as Minuartia stricta (Sw.) Hiern. (Porsild & Cody, 1980; Rabeler, Hartman & Utech, 2005; Elven et al., 2011). A collection reported as Minuartia rossii from the nearby Big Bend area of the Coppermine River (Cody, Reading & Line, 2003) has been re-determined as this species (Reading 5, DAO, det. J. M. Saarela, 2015).

Specimens Examined: Canada. Nunavut: Kitikmeot Region: S of Fockler Creek, along small tributary that runs into Fockler Creek, ca. 2.3 km S of Sandstone Rapids, Coppermine River, 67°25′44.9″N, 115°38′25.9″W ± 100 m, 152 m, 3 July 2014, Saarela, Sokoloff & Bull 3271 (CAN); tundra just S of Fockler Creek and N of unnamed tributary, ca. 2.2 km S of Sandstone Rapids, Coppermine River, 67°25′49″N, 115°38′8.9″W ± 3 m, 152 m, 3 July 2014, Saarela, Sokoloff & Bull 3315 (CAN); NW-facing slope above tributary of Fockler Creek, ca. 2.4 km SSW of Sandstone Rapids, Coppermine River, 67°25′46″N, 115°38′49.4″W ± 50 m, 149 m, 3 July 2014, Saarela, Sokoloff & Bull 3300a (CAN); Sleigh Creek, near its confluence with small, unnamed tributary, ca. 1 km SE of Sandstone Rapids, Coppermine River, 67°27′2″N, 115°37′28.7″W ± 10 m, 150 m, 6 July 2014, Saarela, Sokoloff & Bull 3466 (CAN); confluence of Coppermine River and Melville Creek, just W of Coppermine Mountains, 67°15′52″N, 115°30′55.3″W ± 350 m, 178–190 m, 7 July 2014, Saarela, Sokoloff & Bull 3517 (CAN).

Sagina nodosa subsp. borealis G. E. Crow, Fig. 52—Northern knotted pearlwort | North American (N)–amphi-Atlantic–European (N)–Asian (N) | Noteworthy Record

Figure 52 Sagina nodosa subsp. borealis.

(A) Inflorescence, Saarela et al. 4413. (B) Habit, Saarela et al. 4413. Photographs by P. C. Sokoloff.

Our collections from Kugluk (Bloody Falls) Territorial Park and Kugluktuk are the first records of this shoreline species from the study area, and extend its known northeast from the nearest collections from eastern Great Bear Lake (Porsild, 1943; Crow, 1978; Porsild & Cody, 1980). Recorded from a few Arctic sites elsewhere on mainland Nunavut, southern Baffin Island, islands in Hudson Bay and northern Quebec (Crow, 1978; Porsild & Cody, 1980; Blondeau, 2015a). Porsild & Cody (1980) did not recognise infraspecific taxa in Sagina nodosa (L.) Fenzl, whereas in other treatments two are recognised: subsp. nodosa, a non-Arctic coastal taxon from northeastern North America, and subsp. borealis (Crow, 1978, 2005; Elven et al., 2011). We follow these latter treatments.

Specimens Examined: Canada. Nunavut: Kitikmeot Region: Kugluk (Bloody Falls) Territorial Park, rocky sandy beach just below Bloody Falls, W side of Coppermine River, vicinity of confluence with small creek, beach seasonally flooded, 67°44′54.5″N, 115°22′17.2″W ± 75 m, 9 m, 17 July 2014, Saarela, Sokoloff & Bull 4134 (CAN); grassy sandy flats on extensive sandy floodplain of Coppermine River, below steep cliff above river and S of Kugluktuk, 67°48′54.3″N, 115°6′9.1″W ± 20 m, 5 m, 26 July 2014, Saarela, Sokoloff & Bull 4413 (CAN).

Silene acaulis (L.) Jacq., Fig. S25A—Moss campion | Amphi-Beringian–North American–amphi-Atlantic–European (N/C)–Asian (NW)

Previously recorded from Bloody Falls and Kugluktuk (Cody, 1954b; Porsild & Cody, 1980). We made collections at Fockler Creek, Big Creek, Coppermine Mountains and Kugluk (Bloody Falls) Territorial Park. Widespread throughout the Canadian Arctic (Porsild & Cody, 1980; Cody, Scotter & Zoltai, 1989; Korol, 1992; Aiken et al., 2007; Saarela et al., 2013a; Blondeau, 2015a). Two infraspecific taxa have been recognised, subsp. acaulis and subsp. subacaulescens (F. N. Williams) Hultén (Porsild & Cody, 1980, who recognised them provisionally; Elven et al., 2011). Morton (2005b) does not recognise these, noting considerable intergradation. We follow the latter treatment.

Specimens Examined: Canada. Nunavut: Kitikmeot Region: Coppermine River, Fort Hearne–Bloody Falls [67.7761°N, 115.2037°W ± 7.5 km], 1931, A. M. Berry 4 (CAN-54391); Coppermine [Kugluktuk], Coronation Gulf, at mouth of Coppermine River, ledge of north-facing cliff, hill back of the village [67.822146°N, 115.078387°W ± 0.5 km], 4 August 1948, H. T. Shacklette 3321 (CAN-200286); Bloody Falls on Coppermine River, 67°44′N, 115°23′W, 18 July 1951, W. I. Findlay 141 (DAO-23244 01-01000616635); Coppermine [Kugluktuk], 67°49′36″N, 115°5′36″W, 3 July 1951, W. I. Findlay 82 (DAO-23243 01-01000616636); Kugluktuk, rocky slopes of North Hill, 67°49′29.6″N, 115°6′31″W ± 50 m, 50 m, 29 June 2014, Saarela, Sokoloff & Bull 3055 (CAN, UBC); flats on W side of Fockler Creek, above spruce forest in creek valley, ca. 2.2 km S of Sandstone Rapids, Coppermine River, 67°25′49″N, 115°37′55″W ± 50 m, 152 m, 1 July 2014, Saarela, Sokoloff & Bull 3119 (ALA, CAN); old riverbed of Fockler Creek, ca. 2.3 km SSE of Sandstone Rapids, Coppermine River, 67°25′45.7″N, 115°37′21.8″W ± 25 m, 166 m, 1 July 2014, Saarela, Sokoloff & Bull 3160 (CAN); forest and slopes at confluence of Big Creek and Coppermine River, N side of Coppermine River, S side of Coppermine Mountains, 67°14′29.3″N, 116°2′44.5″W ± 250 m, 180–199 m, 7 July 2014, Saarela, Sokoloff & Bull 3555 (CAN, O); flats atop and upper slopes of Coppermine Mountains, N/W side of Coppermine River, 67°14′49.9″N, 115°38′43.7″W ± 200 m, 467 m, 9 July 2014, Saarela, Sokoloff & Bull 3769 (CAN); Kugluk (Bloody Falls) Territorial Park, upper ledges of rocky (gabbro) S-facing cliffs above the start of Bloody Falls (W bank of River), just E of Portage Trail, 67°44′21.7″N, 115°22′42.2″W ± 25 m, 46 m, 14 July 2014, Saarela, Sokoloff & Bull 3951 (CAN, MT).

Silene involucrata subsp. tenella (Tolm.) Bocquet—Arctic bladder campion | European (NE)–Asian (N)–amphi-Beringian

Recorded previously from Kugluktuk (Porsild & Cody, 1980). We made collections at Fockler Creek, Coppermine Mountains and Kugluk (Bloody Falls) Territorial Park. Neither Aiken et al. (2007) nor Porsild & Cody (1980) recognised infraspecific taxa in Silene involucrata (Cham. & Schltdl.) Bocquet, whereas Elven et al. (2011) and Morton (2005b) did, but there is disagreement as to which subspecies names are correct. We follow Morton (2005b); his subsp. tenella corresponds to subsp. involucrata in Elven et al. (2011). The geographic summary above corresponds to the latter name in Elven et al. (2011). Porsild & Cody (1980) treated the species as Melandrium affine (J. Vahl ex Fries) J. Vahl, now considered a synonym of this species. Melandrium Rohl. is included within Silene L. by all recent authors. Two collections previously reported as Lychnis triflora R. Br. (=Silene sorensenis (B. Boivin) Bocquet) (Findlay 77, 94; Cody, 1954b) were re-determined by G. Bocquet in 1965 as this species, and we agree.

Specimens Examined: Canada. Nunavut: Kitikmeot Region: Coppermine [Kugluktuk], 67°49′36″N, 115°5′36″W, 24 July 1951, W. I. Findlay 177 (DAO-9409 01-01000679540) (Cody, 1954b), 5 July 1951, W. I. Findlay 94 (DAO-9407 01-01000616632); flats on W side of Fockler Creek, above spruce forest in creek valley, ca. 2.2 km S of Sandstone Rapids, Coppermine River, 67°25′49″N, 115°37′55″W ± 50 m, 152 m, 1 July 2014, Saarela, Sokoloff & Bull 3108 (CAN); flats atop and upper slopes of Coppermine Mountains, N/W side of Coppermine River, 67°14′43.7″N, 115°38′51.2″W ± 150 m, 422 m, 9 July 2014, Saarela, Sokoloff & Bull 3745 (CAN); Kugluk (Bloody Falls) Territorial Park, rocky cliffs and ledges directly above (W side) of Bloody Falls, just S of heavily used day-use/fishing area, 67°44′40.1″N, 115°22′4.9″W ± 20 m, 8 m, 12 July 2014, Saarela, Sokoloff & Bull 3809 (CAN).

Silene uralensis (Rupr.) Bocquet subsp. uralensis, Fig. S25B—Mountain campion, nodding campion | European (NE)–Asian (N)–amphi-Beringian–North American (N)

Previously recorded from Kugluktuk (Cody, 1954b, as Lychnis apetala var. arctica (Fr.) Cody; Porsild & Cody, 1980, as Melandrium apetalum (L.) Fenzl). We made collections at Fockler Creek, along the Coppermine River, Kugluk (Bloody Falls) Territorial Park and near Heart Lake. Taxonomic treatments of this species vary considerably. This taxon is treated as Melandrium apetalum subsp. arcticum (Fr.) Hultén in Porsild & Cody (1980), in a broader sense than recognised here by including Silene uralensis subsp. arctica (Fr.) Bocquet, but distinct from the cordilleran taxon Melandrium apetalum subsp. attenuatum (Farr) H. Hara. Morton (2005b) included subsp. arctica and subsp. attenuata in a more broadly circumbscribed Silene uralensis subsp. uralensis. We follow Elven et al. (2011), who recognised three subspecies: subsp. uralensis (with Melandrium apetalum subsp. attenuatum as a tentative synonym), subsp. arctica, a high Arctic taxon, and subsp. ogilviensis (A. E. Porsild) D. F. Brunt., a problematic taxon also recognised by Morton (2005b). Silene uralensis subsp. arctica is not recorded from the study area, but reaches the mainland in Tuktut Nogait National Park and adjacent Nunavut (Saarela et al., 2013a). As well, two collections from west of the Coppermine River outside the study area (Reading 22, 28, DAO) previously determined as subsp. uralensis, are subsp. arctica (det. J. M. Saarela 2015). The distribution of subsp. uralensis sensu Elven et al. (2011) has not been mapped and is therefore unclear, but it was not recorded for the Canadian Arctic Archipelago by Aiken et al. (2007). Silene uralensis s.l., however, is widespread in the Canadian Arctic (Porsild & Cody, 1980; Korol, 1992; Aiken et al., 2007; Saarela et al., 2013a; Blondeau, 2015a).

Specimens Examined: Canada. Nunavut: Kitikmeot Region: Coppermine [Kugluktuk], vicinity of post [67°49′36″N, 115°5′36″W ± 1.5 km], 26 July 1949, A. E. Porsild 17174 (CAN-127723); Coppermine [Kugluktuk], 67°49′36″N, 115°5′36″W, 20 July 1951, W. I. Findlay 155 (ACAD-30917, DAO-9401 01-01000616610, UBC-V40799) (Porsild & Cody, 1980; Korol, 1992; Aiken et al., 2007; Saarela et al., 2013a; Blondeau, 2015a), 8 July 1951, W. I. Findlay 95 (DAO-9402 01-01000616611); slopes on E side of Coppermine River, N of its confluence with Fockler Creek, ca. 0.8 km SW of Sandstone Rapids, 67°26′36.9″N, 115°38′50.1″W ± 50 m, 128 m, 4 July 2014, Saarela, Sokoloff & Bull 3389 (CAN, UBC); esker on E side of Coppermine River, 0.6 km SSE of Muskox Rapids, 67°22′40″N, 115°42′38.5″W ± 50 m, 172 m, 7 July 2014, Saarela, Sokoloff & Bull 3609 (CAN); S-facing slopes on W side of Coppermine River, about halfway between Escape Rapids and Muskox Rapids, 67°31′18.2″N, 115°36′20.1″W ± 150 m, 115 m, 8 July 2014, Saarela, Sokoloff & Bull 3623 (CAN); Kugluk (Bloody Falls) Territorial Park, wet meadow between Coppermine River and large sand hills on W side of river, 0.5 km W of Bloody Falls, 67°44′44.8″N, 115°22′48.3″W ± 15 m, 33 m, 15 July 2014, Saarela, Sokoloff & Bull 4050 (CAN); ca. 0.5 km SW of Heart Lake, SW of Kugluktuk, 7.5 km SW of mouth of Coppermine River, 67°47′52″N, 115°14′14.4″W ± 350 m, 66 m, 23 July 2014, Saarela, Sokoloff & Bull 4290 (ALA, CAN).

Stellaria borealis Bigelow subsp. borealis—Boreal starwort | North American–amphi-Atlantic–European (N) | Noteworthy Record

Newly recorded for the study area, and our single collection, from the Fockler Creek area, represents a range extension from the nearest known sites along eastern Great Bear Lake. Petals are absent in our collection, which was gathered on a wet, lower bank of a snowbed area just above a small creek, where the species was uncommon, growing with Anemone parviflora, Arctous rubra, Carex podocarpa, Cassiope tetragona, Dasiphora fruticosa and Salix reticulata. Porsild & Cody (1980) treated the taxon as Salix calycantha (Ledeb.) Bong. s.l., a broadly distributed circumpolar, non-Arctic species, whereas Morton & Rabeler (1989) recognised two species in the Salix calycantha complex, Salix calycantha s.s. (a non-Arctic western North American taxon) and Salix borealis with two infraspecific taxa: subsp. borealis, a widespread boreal and cordilleran taxon, and subsp. sitchana (Steud.) Piper, a non-Arctic western North American taxon. This latter approach was adopted in subsequent treatments (Morton, 2005c; Elven et al., 2011), which we follow. Elsewhere in Nunavut recorded from Daring Lake, the southeastern mainland, the Yathkyed Lake area, the Boothia peninsula and Akimiski Island (Porsild & Cody, 1980; Riley, 1981; Blaney & Kotanen, 2001; Cody, Reading & Line, 2003; Cody & Reading, 2005). Elsewhere in the Canadian Arctic recorded from the Mackenzie Delta area and northern Quebec and Labrador (Porsild & Cody, 1980; Blondeau, 2015a).

Specimens Examined: Canada. Nunavut: Kitikmeot Region: NW-facing slope above tributary of Fockler Creek, ca. 2.4 km SSW of Sandstone Rapids, Coppermine River, 67°25′46″N, 115°38′49.4″W ± 50 m, 149 m, 3 July 2014, Saarela, Sokoloff & Bull 3290 (CAN).

Stellaria crassifolia Ehrh.—Thick-leaved starwort | Circumboreal-polar | Noteworthy Record

Newly recorded for the study are, extending its range north from the nearby Big Bend area of the Coppermine River (Reading 37, DAO; Cody, Reading & Line, 2003). First collected in the study area near the Kugluktuk Airport in 2013, and we made a single collection in Kugluk (Bloody Falls) Territorial Park. Our collection was made on a sandy, rocky beach where the species was uncommon, growing with Chamerion latifolium, Artemisia tilesii, Equisetum arvense and Salix planifolia. The next-nearest site is eastern Great Bear Lake (Porsild & Cody, 1980). Elsewhere in the Canadian Arctic recorded from Baffin, Prince Charles, Melville and Victoria islands, the Mackenzie Delta area, scattered sites on mainland Nunavut, and northern Quebec and Labrador (Porsild & Cody, 1980; Cody, Reading & Line, 2003; Aiken et al., 2007; Blondeau, 2015a).

Specimens Examined: Canada. Nunavut: Kitikmeot Region: Kugluktuk, airport, 21 July 2013, 67.81749°N, 115.13449°W, B. A. Bennett 13-0634 (CAN); Kugluk (Bloody Falls) Territorial Park, rocky sandy beach just below Bloody Falls, W side of Coppermine River, vicinity of confluence with small creek, beach seasonally flooded, 67°44′54.5″N, 115°22′17.2″W ± 75 m, 9 m, 17 July 2014, Saarela, Sokoloff & Bull 4116 (CAN).

Stellaria humifusa Rottb.—Salt-marsh starwort, low sandwort | Circumpolar–amphi-Pacific

Previously recorded for the Kugluktuk area (Porsild & Cody, 1980), but we were unable to locate a voucher. We made collections of this shoreline species along Richardson Bay and at Heart Lake. Widespread throughout the Canadian Arctic Archipelago and known from several mainland Arctic sites in Canada (Porsild & Cody, 1980; Korol, 1992; Aiken et al., 2007; Saarela et al., 2013a; Blondeau, 2015a).

Specimens Examined: Canada. Nunavut: Kitikmeot Region: Richardson Bay, confluence of Richardson and Rae rivers at Coronation Gulf, ca. 20 km WNW of Kugluktuk, 67°54′11.2″N, 115°32′27.4″W ± 200 m, 0 m, 8 July 2014, Saarela, Sokoloff & Bull 3673 (ALA, ALTA, CAN, O); Heart Lake, SW of Kugluktuk, 6.4 km SW of mouth of Coppermine River, 67°48′7.8″N, 115°13′22.7″W ± 350 m, 33 m, 23 July 2014, Saarela, Sokoloff & Bull 4293 (CAN, UBC).

Stellaria longipes Goldie—Long-stalked starwort, Goldie’s starwort | Circumboreal–polar

Previously recorded from Kugluktuk, as Stellaria ciliatosepala Trautv. (Cody, 1954b) and Stellaria monantha Hultén var. monantha (Porsild & Cody, 1980). We made collections at Fockler Creek, Coppermine Mountains, Kugluk (Bloody Falls) Territorial Park, Heart Lake and Kugluktuk. Widespread throughout the Canadian Arctic (Porsild & Cody, 1980; Cody, Scotter & Zoltai, 1984, 1989; Korol, 1992; Cody & Reading, 2005; Aiken et al., 2007; Saarela et al., 2013a; Blondeau, 2015a). Porsild & Cody (1980) recognised seven species as part of the Stellaria longipes species complex (Stellaria arenicola Raup, Stellaria stricta Richardson, Stellaria subvestita Greene, Stellaria crassipes Hultén, Stellaria monantha, Stellaria edwardsii R. Br., Stellaria laeta Richardson) based on the earlier treatment of Porsild (1963). Elven et al. (2011) provisionally recognised these species as unranked infraspecific “taxa”. Other authors recognise a single polymorphic taxon, with most of the above-named taxa included in Stellaria longipes subsp. longipes. The other infraspecific taxon is the non-Arctic subsp. arenicola (Raup) C. C. Chinnappa & J. K. Morton from the Lake Athabasca sand dunes in Saskatchewan and Alberta (recognised as Stellaria arenicola Raup in Porsild & Cody, 1980) (Chinnappa & Morton, 1991; Morton, 2005c). We follow the latter approach.

Specimens Examined: Canada. Nunavut: Kitikmeot Region: Coppermine River, Fort Hearne–Bloody Falls [67.7761972°N, 115.2037222°W ± 7.5 km], 1931, A. M. Berry 5 (CAN-51226); Coppermine [Kugluktuk], 67°49′36″N, 115°5′36″W, 10 July 1951, W. I. Findlay 111 (DAO-187221 01-01000617008); Coppermine [Kugluktuk], common above Eskimo [sic] tents west of DOT [67.826667°N, 115.09333°W ± 1.5 km], 6 July 1958, R. D. Wood s.n. (CAN-265560); Coppermine [Kugluktuk], in settlement near R.C. [Roman Catholic] Mission [67.826667°N, 115.09333°W ± 1.5 km], 8 July 1958, R. D. Wood s.n. (CAN-265561); Coppermine [Kugluktuk], vic. of hamlet and airstrip, 67.78°N, 115.5°W ± 3,615 m, 23 June 1999, C. L. Parker & I. Jonsdottir 9102 (ALA); Kugluktuk, disturbed rocky/gravelly ground around house, top of NW-facing gravel slope, just SE of Igalik Building, 67°49′29.6″N, 115°6′31″W ± 10 m, 13 m, 30 June 2014, Saarela, Sokoloff & Bull 3097 (CAN); flats on W side of Fockler Creek, above spruce forest in creek valley, ca. 2.2 km S of Sandstone Rapids, Coppermine River, 67°25′49″N, 115°37′55″W ± 50 m, 152 m, 1 July 2014, Saarela, Sokoloff & Bull 3133 (CAN); S of Fockler Creek, along small tributary that runs into Fockler Creek, ca. 2.3 km S of Sandstone Rapids, Coppermine River, 67°25′44.9″N, 115°38′25.9″W ± 100 m, 152 m, 3 July 2014, Saarela, Sokoloff & Bull 3249 (CAN); E side of Fockler Creek, in valley just above creek’s confluence with the Coppermine River, ca. 1.4 km SSW of Sandstone Rapids, 67°26′21.4″N, 115°38′54″W ± 5 m, 140 m, 4 July 2014, Saarela, Sokoloff & Bull 3361 (CAN, MO); flats atop and upper slopes of Coppermine Mountains, N/W side of Coppermine River, 67°14′49.9″N, 115°38′43.7″W ± 200 m, 467 m, 9 July 2014, Saarela, Sokoloff & Bull 3767 (CAN); Kugluk (Bloody Falls) Territorial Park, rocky cliffs and ledges directly above (W side) of Bloody Falls, just S of heavily used day-use/fishing area, 67°44′40.1″N, 115°22′4.9″W ± 20 m, 8 m, 12 July 2014, Saarela, Sokoloff & Bull 3812 (CAN, MT); SSW-facing slopes above start of Bloody Falls, SE side of Coppermine River, 67°44′12.5″N, 115°22′31″W ± 50 m, 50–60 m, 19 July 2014, Saarela, Sokoloff & Bull 4207 (CAN, US); W of Kugluktuk on tundra flats above Coppermine River, S of 1 Coronation Drive and N of community power plant, 67°49′28.97″N, 115°5′0.2″W ± 100 m, 8 m, 22 July 2014, Saarela, Sokoloff & Bull 4239 (CAN); Heart Lake, SW of Kugluktuk, 6.4 km SW of mouth of Coppermine River, 67°48′7.8″N, 115°13′22.7″W ± 350 m, 33 m, 23 July 2014, Saarela, Sokoloff & Bull 4341 (CAN); Kugluktuk, roadside and flats between buildings, 67°49′27.4″N, 115°5′26.2″W ± 25 m, 29 m, 26 July 2014, Saarela, Sokoloff & Bull 4394 (ALA, CAN).

Chenopodiaceae [1/1]

Suaeda calceoliformis (Hook.) Moq., Fig. 53—Horned sea-blite | North American

Figure 53 Suaeda calceoliformis.

(A) Habit, Saarela et al. 3668. (B) Habit, Saarela et al. 3668. (C) Habitat, Saarela et al. 3668. Photographs by R. D. Bull.

This halophytic species has been recorded from the same general area as our collection (Bassett & Crompton, 1978, as “Suaeda depressa sensu S. Wats”.; Porsild & Cody, 1980; Ferren & Schenk, 2004), but has not been found elsewhere in the study area. We found it to be locally common and forming dense patches on wet and muddy mid- to upper-slopes of estuary channels at Richardson Bay, growing with Carex subspathacea, Potentilla anserina, Puccinellia phryganodes and Stellaria humifusa. Elsewhere in the Canadian Arctic recorded from the Kent Peninsula in Nunavut (Cody, Reading & Line, 2003) and Victoria Island, the first records for the Canadian Arctic Archipelago (Gillespie et al., 2015).

Specimens Examined: Canada. Nunavut: Kitikmeot Region: Rae River, mouth, 2 August 1955, R. E. Miller 312 (CAN-241966); Richardson Bay, confluence of Richardson and Rae rivers at Coronation Gulf, ca. 20 km WNW of Kugluktuk, 67°54′11.2″N, 115°32′27.4″W ± 200 m, 0 m, 8 July 2014, Saarela, Sokoloff & Bull 3668 (CAN).

Elaeagnaceae [1/1]

Shepherdia canadensis (L.) Nutt., Fig. 54—Canada buffaloberry, soapberry, foamberry | North American | Noteworthy Record

Figure 54 Shepherdia canadensis.

(A) Fruits, Saarela et al. 3148. (B) Habit, Saarela et al. 3148. (C) Habitat, Saarela et al. 3148. Photographs by R. D. Bull.

Newly recorded for the study area and slight northern range extension from the nearby Big Bend area of the Coppermine River (Reading 26, DAO; Cody & Reading, 2005). The next-nearest known sites are along the eastern arm of Great Bear Lake (Porsild, 1943; Porsild & Cody, 1980). We made collections at Fockler Creek, Big Creek, sites along the Coppermine River and Kugluk (Bloody Falls) Territorial Park. This is a fairly common shrub in the study area that grows along the banks of the Coppermine River and in sheltered areas of the park. We recorded it as far north as the park, but it may extend further north along the Coppermine River. It is surprising that previous collectors at Bloody Falls did not collect this conspicuous woody species. Elsewhere in Nunavut known from the lower Hood River (Gould & Walker, 1997) (mostly at their sites one to four), the southeastern mainland (Cody & Reading, 2005), Akimiski Island (Blaney & Kotanen, 2001) and the Belcher Islands (Consaul et al. 4230 (CAN-599700), 3817 (CAN-599607), 4259 (CAN-599701), 4253 (CAN-599702), 3723 (CAN-599605), Consaul 4003 (CAN-5996606)). Elsewhere in the Canadian Arctic known from the Brock River, Mackenzie Delta area, northern Yukon and northern Alaska (Porsild & Cody, 1980; Saarela et al., 2013a).

Specimens Examined: Canada. Nunavut: Kitikmeot Region: old riverbed of Fockler Creek, ca. 2.3 km SSE of Sandstone Rapids, Coppermine River, 67°25′48″N, 115°37′33″W ± 25 m, 153 m, 1 July 2014, Saarela, Sokoloff & Bull 3148 (CAN, UBC); S side of Fockler Creek, ca. 2.7 SE of Sandstone Rapids, Coppermine River, 67°25′38.2″N, 115°36′54.9″W ± 50 m, 128 m, 5 July 2014, Saarela, Sokoloff & Bull 3401 (ALA, CAN); forest and slopes at confluence of Big Creek and Coppermine River, N side of Coppermine River, S side of Coppermine Mountains, 67°14′29.3″N, 116°2′44.5″W ± 250 m, 180–199 m, 7 July 2014, Saarela, Sokoloff & Bull 3567 (CAN, UBC, US); S-facing slopes above Coppermine River, ca. 7.8 km NNE of Sandstone Rapids, 67°31′16.2″N, 115°36′52.1″W ± 50 m, 110 m, 8 July 2014, Saarela, Sokoloff & Bull 3643 (ALTA, CAN); SE-facing slopes above Escape Rapids, W side of Coppermine River, 67°36′58.7″N, 115°29′18.3″W ± 99 m, 50 m, 8 July 2014, Saarela, Sokoloff & Bull 3735 (CAN, O); Kugluk (Bloody Falls) Territorial Park, rocky cliffs and ledges directly above (W side) of Bloody Falls, just S of heavily used day-use/fishing area, 67°44′40.1″N, 115°22′4.9″W ± 20 m, 8 m, 12 July 2014, Saarela, Sokoloff & Bull 3818 (CAN, MO); Kugluk (Bloody Falls) Territorial Park, sandy NE-facing slope above small creek in deep gully, about 0.5 km W of Bloody Falls, 67°44′36.6″N, 115°22′59.3″W ± 41 m, 41 m, 15 July 2014, Saarela, Sokoloff & Bull 4016 (CAN, MT).

Ericaceae [10/14]

Andromeda polifolia L., Figs. S26A and S26B—Bog rosemary | Circumboreal–polar

Previously recorded from Kugluktuk (Macoun & Holm, 1921; Cody, 1954b; Porsild & Cody, 1980). We made collections at Fockler Creek, Kendall River and Kugluk (Bloody Falls) Territorial Park. This low shrub was previously considered to be restricted to the mainland, but was recently reported from Victoria Island and southern Baffin Island (Gillespie et al., 2015). There are numerous mainland Arctic records in Canada (Porsild & Cody, 1980; Cody, Scotter & Zoltai, 1989; Korol, 1992; Saarela et al., 2013a).

Specimens Examined: Canada. Nunavut: Kitikmeot Region: Coppermine [Kugluktuk], 67°49′36″N, 115°5′36″W, 8 July 1951, W. I. Findlay 104 (ACAD-30951); Coppermine [Kugluktuk], vicinity of post [67°49′36″N, 115°5′36″W ± 1.5 km], 26 July 1949, A. E. Porsild 17182 (CAN-128044); Coppermine [Kugluktuk], 67°49′36″N, 115°5′36″W, 23 June 1951, W. I. Findlay 29 (DAO-181418 01-000002656); Coppermine [Kugluktuk], 67°51′N, 115°16′W, 2 July 1972, F. Fodor N 137 (UBC-V151895); Kugluktuk, upper tundra slope overlooking Coppermine River [67°49′36″N, 115°5′36″W], 16 July 2000, L. K. Benjamin s.n. (ACAD-ECS015866); Kugluktuk, rocky slopes of North Hill, 67°49′29.6″N, 115°6′31″W ± 50 m, 50 m, 29 June 2014, Saarela, Sokoloff & Bull 3049 (CAN, UBC); ridge N of Fockler Creek, ca. 2.1 km SSE of Sandstone Rapids, Coppermine River, 67°25′54″N, 115°37′30″W ± 25 m, 166 m, 2 July 2014, Saarela, Sokoloff & Bull 3208 (ALA, CAN); confluence of Coppermine and Kendall rivers (NW side of Coppermine River, S side of Kendall River), 67°6′51.1″N, 116°8′18.3″W ± 150 m, 220 m, 7 July 2014, Saarela, Sokoloff & Bull 3592 (CAN); Kugluk (Bloody Falls) Territorial Park, rocky valley immediately SW of Bloody Falls, along rough marked section of Portage Trail, 67°44′34″N, 115°22′16″W ± 50 m, 20 m, 13 July 2014, Saarela, Sokoloff & Bull 3883 (ALA, CAN, MT).

Arctostaphylos uva-ursi (L.) Spreng.—Common bearberry, kinnikinnick | Circumboreal | Noteworthy Record

Newly recorded for the study area. Our single Subarctic collection, gathered from low shrub tundra in a white spruce forest along Melville Creek, represents a northeastern range extension from the eastern shore of Great Bear Lake (Porsild & Cody, 1980). Elsewhere in Nunavut known from the Nueltin Lake area of the southeastern mainland (Porsild, 1950b; Porsild & Cody, 1980) and Akimiski Island (Blaney & Kotanen, 2001). It reaches the Arctic in North America in western Greenland, Northwest Territories (e.g., Richards Island, Cody & Gutteridge 7963, ALTA-19107, not seen), Yukon and northern Alaska (Hultén, 1971; Porsild & Cody, 1980; Bennett, 2008), but there are no Arctic records from Nunavut. Infraspecific taxa are not currently recognised in the species (Parker, Vasey & Keeley, 2009; Elven et al., 2011).

Specimens Examined: Canada. Nunavut: Kitikmeot Region: confluence of Coppermine River and Melville Creek, just W of Coppermine Mountains, 67°15′52″N, 115°30′55.3″W ± 350 m, 178–190 m, 7 July 2014, Saarela, Sokoloff & Bull 3534 (CAN).

Arctous alpina (L.) Nied.—Alpine bearberry | Circumpolar-alpine

Previously recorded from Kugluktuk, as Arctostaphylos alpina (L.) Spreng. (Cody, 1954b; Porsild & Cody, 1980), where we also made collections. Elsewhere in the Canadian Arctic recorded from southern Baffin Island, Banks, Southampton and Victoria islands, and numerous mainland sites (Porsild & Cody, 1980; Cody, Scotter & Zoltai, 1989; Korol, 1992; Cody & Reading, 2005; Aiken et al., 2007). Arctous alpina and Arctous rubra can sometimes be difficult to tell apart, particularly when relying solely on the presence or absence of marcescent leaves. The most reliable character is fruit colour (black in Arctous alpina, red in Arctous rubra), but ripe fruits are rarely available in herbarium material. Most authors recognise these taxa at species level (Porsild & Cody, 1980; Tucker, 2009; Elven et al., 2011), but they have also been treated as varieties (Welsh, 1974). The two can be distinguished by rbcL sequences (Saarela et al., 2013b).

Specimens Examined: Canada. Nunavut: Kitikmeot Region: Coppermine [Kugluktuk], 67°49′36″N, 115°5′36″W, 23 June 1951, W. I. Findlay 29 (DAO-181418 01-000002656); Kugluktuk, 67.821917°N, 115.085033°W, 12 July 2006, J. Davis 621 (CAN-597650); Kugluktuk, rocky slopes of North Hill, 67°49′31.4″N, 115°6′54″W ± 100 m, 42 m, 29 June 2014, Saarela, Sokoloff & Bull 3075 (CAN, UBC); rocky cliffs on S side of Kugluktuk, 67°49′13″N, 115°5′55.8″W ± 50 m, 65 m, 26 July 2014, Saarela, Sokoloff & Bull 4401 (CAN).

Arctous rubra (Rehder & E. H. Wilson) Nakai—Red bearberry | Asian (NE)–amphi-Beringian–North American (N)

Previously recorded Kugluktuk, as Arctostaphylos rubra (Rehder & E. H. Wilson) Fernald (Cody, 1954b; Porsild & Cody, 1980). We made collections at Fockler Creek and Kugluk (Bloody Falls) Territorial Park. Elsewhere in the Canadian Arctic recorded from Banks, Baffin (one record that should be confirmed), Southampton and Victoria islands, and a few mainland sites (Porsild & Cody, 1980; Aiken et al., 2007; Saarela et al., 2013a). See additional comments under Arctous alpina.

Specimens Examined: Canada. Nunavut: Kitikmeot Region: Coppermine [Kugluktuk], 67°49′36″N, 115°5′36″W, 6 June 1951, W. I. Findlay 3 (DAO-181442 01-000003622); old riverbed of Fockler Creek, ca. 2.3 km SSE of Sandstone Rapids, Coppermine River, 67°25′45.7″N, 115°37′21.8″W ± 25 m, 166 m, 1 July 2014, Saarela, Sokoloff & Bull 3167 (CAN, UBC); NW-facing slope just upstream of small tributary from its confluence with Fockler Creek, ca. 2.4 km SSW of Sandstone Rapids, Coppermine River, 67°25′46″N, 115°38′49.4″W ± 200 m, 149 m, 3 July 2014, Saarela, Sokoloff & Bull 3313 (ALA, CAN); Kugluk (Bloody Falls) Territorial Park, rocky cliffs and ledges directly above (W side) of Bloody Falls, just S of heavily used day-use/fishing area, 67°44′40.1″N, 115°22′4.9″W ± 20 m, 8 m, 12 July 2014, Saarela, Sokoloff & Bull 3825 (ALTA, CAN).

Cassiope tetragona (L.) Don. subsp. tetragona, Fig. S26C—Arctic heather | Circumpolar–alpine

Previously recorded from Bloody Falls and Kugluktuk (Macoun & Holm, 1921; Cody, 1954b; Porsild & Cody, 1980). We made collections of this common snowbed species at Fockler Creek and Kugluk (Bloody Falls) Territorial Park. Known from throughout the Canadian Arctic (Porsild & Cody, 1980; Cody, Scotter & Zoltai, 1989; Korol, 1992; Aiken et al., 2007; Saarela et al., 2013a). The other infraspecific taxon, subsp. saximontana (Small) A. E. Porsild, is non-Arctic (Wallace, 2009).

Specimens Examined: Canada. Nunavut: Kitikmeot Region: Bloody Falls, 67°44′N, 115°23′W, 18 July 1951, W. I. Findlay 137 (DAO-113610 01-000003422); Coppermine [Kugluktuk], 67°49′36″N, 115°5′36″W, 29 June 1951, W. I. Findlay 49 (ACAD-30952, DAO-113611 01-000003421, UBC-V40788); Coppermine [Kugluktuk], 67°51′N, 115°16′W, 2 July 1972, F. Fodor N 146 (UBC-V151907); Kugluktuk, 67.82445°N, 115.169783°W, 21 June 2006, J. Davis 602 (CAN-597640); Coppermine River, Fort Hearne–Bloody Falls [67.7761972°N, 115.2037222°W ± 7.5 km], 1931, A. M. Berry 19 (CAN-88524); Kugluktuk, rocky outcrop, overlooking Coppermine River [67°49′36″N, 115°5′36″W], 27 June 2000, L. K. Benjamin s.n. (ACAD-ECS015879); Kugluktuk, rocky slopes of North Hill, 67°49′29.6″N, 115°6′31″W ± 50 m, 50 m, 29 June 2014, Saarela, Sokoloff & Bull 3061 (ALA, ALTA, CAN); NW-facing slope above tributary of Fockler Creek, ca. 2.4 km SSW of Sandstone Rapids, Coppermine River, 67°25′46″N, 115°38′49.4″W ± 50 m, 149 m, 3 July 2014, Saarela, Sokoloff & Bull 3289 (CAN, UBC); Kugluk (Bloody Falls) Territorial Park, rocky valley immediately SW of Bloody Falls, along rough marked section of Portage Trail, 67°44′34″N, 115°22′16″W ± 50 m, 20 m, 13 July 2014, Saarela, Sokoloff & Bull 3886 (CAN, MO, O).

Empetrum nigrum L., Figs. S27A and S27B—Crowberry, blackberry | Circumboreal–polar

Previously recorded from Kugluktuk (Cody, 1954b; Porsild & Cody, 1980). We made collections at Fockler Creek, Melville Creek and Kugluk (Bloody Falls) Territorial Park. Elsewhere in the Canadian Arctic known from Baffin, Ellesmere, Southampton and Victoria islands, and numerous mainland sites (Porsild & Cody, 1980; Cody, Scotter & Zoltai, 1989; Korol, 1992; Aiken et al., 2007; Saarela et al., 2013a). Taxonomy follows Murray, Mirré & Elven (2008) and Elven et al. (2011), who recognise the polymorphic taxon in a broad sense. Porsild & Cody (1980) recognised North American plants as Empetrum nigrum subsp. hermaphroditum (Lge.) Böcher.

Specimens Examined: Canada. Nunavut: Kitikmeot Region: Coppermine [Kugluktuk], 67°49′36″N, 115°5′36″W, W. I. Findlay 35 (DAO-180513) Porsild & Cody (1980), W. I. Findlay 48 (DAO-180510); Kugluktuk, 67.81875°N, 115.0885°W, 10 July 2006, J. Davis 620 (CAN-597642); NW-facing slope just upstream of small tributary from its confluence with Fockler Creek, ca. 2.4 km SSW of Sandstone Rapids, Coppermine River, 67°25′46″N, 115°38′49.4″W ± 200 m, 149 m, 3 July 2014, Saarela, Sokoloff & Bull 3311 (ALTA, CAN, O); confluence of Coppermine River and Melville Creek, just W of Coppermine Mountains, 67°15′52″N, 115°30′55.3″W ± 350 m, 178–190 m, 7 July 2014, Saarela, Sokoloff & Bull 3533 (CAN, UBC); Kugluk (Bloody Falls) Territorial Park, rocky cliffs and ledges directly above (W side) of Bloody Falls, just S of heavily used day-use/fishing area, 67°44′40.1″N, 115°22′4.9″W ± 20 m, 8 m, 12 July 2014, Saarela, Sokoloff & Bull 3824 (ALA, CAN).

Kalmia procumbens (L.) Gift & Kron ex Galasso, Banfi & F. Conti—Alpine azalea | Asian (NE)–amphi-Beringian–North American (N)–amphi-Atlantic–European (N)

Previously recorded from the study area, as Loiseleuria procumbens (L.) Desv. (Porsild & Cody, 1980), but we were unable to locate a specimen for confirmation. We encountered this rare species only at the Bigtree River site where it was uncommon in a dry spruce forest among granite/gabbro outcrops growing with Betula glandulosa and Vaccinium uliginosum. Elsewhere in the Canadian Arctic known from southern Baffin Island and a few mainland sites (Porsild & Cody, 1980; Korol, 1992; Aiken et al., 2007; Saarela et al., 2013a). It is at the northern edge of its known range in the study area.

Specimens Examined: Canada. Nunavut: Kitikmeot Region: confluence of Coppermine and Bigtree rivers, 66°56′23.8″N, 116°21′3.2″W ± 100 m, 265 m, 7 July 2014, Saarela, Sokoloff & Bull 3608 (CAN).

Orthilia secunda subsp. obtusata (Turcz.) Böcher, Figs. S27C and S27D—One-sided wintergreen, nodding wintergreen | Asian (N/C)–amphi-Beringian–North American

Previously recorded from Kugluktuk (Cody, 1954b; Porsild & Cody, 1980). We made collections at Sandstone Rapids, Kendall River, Coppermine Mountains, Kugluk (Bloody Falls) Territorial Park and Kugluktuk. Orthilia secunda s.l. (including Orthilia obtusata Turcz.) is a circumpolar taxon. In North America two (Porsild & Cody, 1980) or no (Cody, 2000; Freeman, 2009a) infraspecific taxa have been recognised, while in a Panarctic context the taxa are tentatively recognised as species (Elven et al., 2011) with an acknowledgement that in northwestern North America they can be difficult to distinguish. We take an intermediate approach and recognise the taxa as subspecies. All of our collections are the Arctic-alpine race with small (<2 cm long) elliptic to orbicular leaf blades. The map in Porsild & Cody (1980) does not distinguish between the infraspecific taxa, while Aiken et al. (2007) mapped this subspecies in the Canadian Arctic from Victoria Island and a few mainland sites. We recorded this subspecies from Tuktut Nogait National Park and vicinity (Saarela et al., 2013a). Orthilia secunda was previously classified as Pyrola secunda L.

Specimens Examined: Canada. Nunavut: Kitikmeot Region: Coppermine [Kugluktuk], 67°49′36″N, 115°5′36″W, 4 August 1951, W. I. Findlay 250 (DAO-32677 01-01000619901); Coppermine River, sandstone cliffs above Sandstone Rapids, 67°27′29.6″N, 115°37′59.3″W ± 100 m, 110 m, 6 July 2014, Saarela, Sokoloff & Bull 3479 (CAN); confluence of Coppermine and Kendall rivers (NW side of Coppermine River, S side of Kendall River), 67°6′51.1″N, 116°8′18.3″W ± 150 m, 220 m, 7 July 2014, Saarela, Sokoloff & Bull 3584 (CAN); S-facing slopes above Coppermine River, ca. 7.8 km NNE of Sandstone Rapids, 67°31′16.2″N, 115°36′52.1″W ± 50 m, 110 m, 8 July 2014, Saarela, Sokoloff & Bull 3640 (CAN); flats atop and upper slopes of Coppermine Mountains, N/W side of Coppermine River, 67°36′58.7″N, 115°29′18.3″W ± 99 m, 50 m, 8 July 2014, Saarela, Sokoloff & Bull 3784 (CAN); Kugluk (Bloody Falls) Territorial Park, SE-facing slope above small stream in deep gully that runs into Coppermine River just below Bloody Falls, ca. 1 km W of Bloody Falls, 67°44′41.2″N, 115°23′34.8″W ± 50 m, 49 m, 15 July 2014, Saarela, Sokoloff & Bull 4037 (CAN); Kugluk (Bloody Falls) Territorial Park, W side of Coppermine River, between Sandy Hills and Bloody Falls, 67°45′17.6″N, 115°22′14.2″W ± 20 m, 76 m, 17 July 2014, Saarela, Sokoloff & Bull 4139 (CAN, UBC); W of Kugluktuk on tundra flats above Coppermine River, S of 1 Coronation Drive and N of power plant, 67°49′28.97″N, 115°5′0.2″W ± 100 m, 8 m, 25 July 2014, Saarela, Sokoloff & Bull 4385 (ALA, CAN).

Pyrola grandiflora Radius subsp. grandiflora, Fig. S28A—Large-flowered wintergreen | Circumpolar

Previously recorded from Bloody Falls and Kugluktuk (Cody, 1954b; Porsild & Cody, 1980). We made collections at Kugluktuk, Fockler Creek and Kugluk (Bloody Falls) Territorial Park. The three varieties recognised in Porsild & Cody (1980) are not recognised by Freeman (2009b). Elven et al. (2011), whom we follow, recognise two subspecies, of which only subsp. grandiflora is present in North America. Elsewhere in the Canadian Arctic recorded from Baffin, Banks, Coats, Ellesmere, Melville and Southampton islands, and sites across the mainland (Porsild & Cody, 1980; Cody, Scotter & Zoltai, 1989; Korol, 1992; Aiken et al., 2007; Saarela et al., 2013a).

Specimens Examined: Canada. Nunavut: Kitikmeot Region: Bloody Falls, 67°44′N, 115°23′W, 18 July 1951, W. I. Findlay 154 (DAO-22681 01-01000619930); Coppermine [Kugluktuk], 67°49′36″N, 115°5′36″W, 22 July 1951, W. I. Findlay 168 (ACAD-30922, ALTA-VP-13677, DAO-22682 01-01000619931, UBC-V238247); Coppermine [Kugluktuk], 67°49′36″N, 115°5′36″W, 2 August 1995, T. Dolman 96 (LEA); Coppermine [Kugluktuk], vic. of hamlet and airstrip, 67.78°N, 115.5°W ± 3,615 m, 23 June 1999, C. L. Parker & I. Jonsdottir 9098 (ALA); Kugluktuk, upper tundra slope overlooking Coppermine River [67°49′36″N, 115°5′36″W], 16 July 2000, L. K. Benjamin s.n. (ACAD-ECS015873); Kugluktuk, flat mesa at top of North Hill, 67°49′32″N, 115°6′39″W ± 100 m, 50 m, 29 June 2014, Saarela, Sokoloff & Bull 3090 (CAN); old riverbed of Fockler Creek, ca. 2.3 km SSE of Sandstone Rapids, Coppermine River, 67°25′48″N, 115°37′33″W ± 25 m, 153 m, 1 July 2014, Saarela, Sokoloff & Bull 3144 (ALA, ALTA, CAN); Kugluk (Bloody Falls) Territorial Park, rocky cliffs and ledges directly above (W side) of Bloody Falls, just S of heavily used day-use/fishing area, 67°44′40.1″N, 115°22′4.9″W ± 20 m, 8 m, 12 July 2014, Saarela, Sokoloff & Bull 3823 (CAN, UBC).

Rhododendron groenlandicum (Oeder) Kron & Judd—Common Labrador tea | North American (N) | Noteworthy Record

Newly recorded for the study area and a minor northeastern range extension. We made five collections at Fockler Creek, Melville Creek, south of Escape Rapids and Kugluk (Bloody Falls) Territorial Park. This boreal shrub was recently recorded from the nearby Big Bend area of the Coppermine River (Reading 2, DAO; Cody, Reading & Line, 2003), representing the first record for western Nunavut. The next-nearest collections are from the eastern shore of Great Bear Lake (Porsild & Cody, 1980). Elsewhere in Nunavut recorded from the southeastern non-Arctic mainland (Porsild & Cody, 1980), the Arctic mainland (Cody & Reading, 2005) and Akimiski Island (Blaney & Kotanen, 2001). Previously treated in Ledum L. (Ledum groenlandicum Oeder).

Specimens Examined: Canada. Nunavut: Kitikmeot Region: S of Fockler Creek, along small tributary that runs into Fockler Creek, ca. 2.3 km S of Sandstone Rapids, Coppermine River, 67°25′44.9″N, 115°38′25.9″W ± 100 m, 152 m, 3 July 2014, Saarela, Sokoloff & Bull 3250 (CAN, UBC); confluence of Coppermine River and Melville Creek, just W of Coppermine Mountains, 67°15′52″N, 115°30′55.3″W ± 350 m, 178–190 m, 7 July 2014, Saarela, Sokoloff & Bull 3531 (CAN, MO, O); S-facing slopes on W side of Coppermine River, about halfway between Escape Rapids and Muskox Rapids, 67°31′18.2″N, 115°36′20.1″W ± 150 m, 115 m, 8 July 2014, Saarela, Sokoloff & Bull 3621 (ALA, CAN); Kugluk (Bloody Falls) Territorial Park, NE-facing slope of large hill just S of Bloody Falls, W side of Coppermine River, 67°44′6.6″N, 115°23′13.4″W ± 50 m, 40 m, 14 July 2014, Saarela, Sokoloff & Bull 3981 (ALTA, CAN); Kugluk (Bloody Falls) Territorial Park, W side of Coppermine River, just above Bloody Falls, 67°44′22.6″N, 115°22′52″W ± 20 m, 40 m, 16 July 2014, Saarela, Sokoloff & Bull 4107 (CAN).

Rhododendron lapponicum (L.) Wahlenb., Fig. S28B—Lapland rosebay | Asian (NE)–amphi-Beringian–North American (N)–amphi-Atlantic (W)

Previously recorded from Kugluktuk (Macoun & Holm, 1921; Cody, 1954b; Porsild & Cody, 1980). We made collections at Fockler Creek, Kugluktuk and Kugluk (Bloody Falls) Territorial Park. Elsewhere in the Canadian Arctic recorded from Baffin, Banks, Devon, Southampton and Victoria islands, and numerous mainland sites (Porsild & Cody, 1980; Cody, Scotter & Zoltai, 1989; Korol, 1992; Aiken et al., 2007; Saarela et al., 2013a; Bennett, 2015). Elven (2011) provisionally recognise two subspecies, subsp. lapponicum and subsp. alpinum (Glehn) A. P. Khokhr., following Russian treatments, with both reported as occurring in Canada. Since the differences between these putative taxa are unclear in North America, they are not recognised here, as in Judd & Kron (2009) who did not mention them.

Specimens Examined: Canada. Nunavut: Kitikmeot Region: Coppermine [Kugluktuk], 67°49′36″N, 115°5′36″W, 29 June 1951, W. I. Findlay 52 (DAO-181174 01-000001180); Coppermine [Kugluktuk], 67°51′N, 115°16′W, 2 July 1972, F. Fodor N 142 (UBC-V151910); Kugluktuk, rocky slopes of North Hill, 67°49′29.6″N, 115°6′31″W ± 50 m, 50 m, 29 June 2014, Saarela, Sokoloff & Bull 3048 (ALTA, CAN, O); S of Fockler Creek, along small tributary that runs into Fockler Creek, ca. 2.3 km S of Sandstone Rapids, Coppermine River, 67°25′44.9″N, 115°38′25.9″W ± 100 m, 152 m, 3 July 2014, Saarela, Sokoloff & Bull 3245 (CAN, UBC); Kugluk (Bloody Falls) Territorial Park, rocky cliffs and ledges directly above (W side) of Bloody Falls, just S of heavily used day-use/fishing area, 67°44′40.1″N, 115°22′4.9″W ± 20 m, 8 m, 12 July 2014, Saarela, Sokoloff & Bull 3831 (ALA, CAN).

Rhododendron tomentosum subsp. decumbens (Aiton) Elven & D. F. Murray, Fig. S29A—Dwarf Labrador tea | Asian (N/C)–amphi-Beringian–North American (N)

Previously recorded from Bloody Falls and Kugluktuk (Cody, 1954b; Porsild & Cody, 1980). We made collections at Kugluktuk, Fockler Creek and Kugluk (Bloody Falls) Territorial Park. Elsewhere in the Canadian Arctic recorded from Baffin, Southampton and Victoria islands, and numerous mainland sites (Porsild & Cody, 1980; Cody, Scotter & Zoltai, 1989; Korol, 1992; Aiken et al., 2007; Saarela et al., 2013a; Bennett, 2015). Taxonomy follows Elven et al., 2011; the nominate subspecies does not occur in North America. Previously treated in Ledum (Ledum decumbens (Aiton) Lodd. ex Steud. or Ledum palustre var. decumbens Aiton).

Specimens Examined: Canada. Nunavut: Kitikmeot Region: Bloody Falls, 67°44′N, 115°23′W, 18 July 1951, W. I. Findlay 139 (DAO-181111 01-01000619934); Coppermine [Kugluktuk], 67°49′36″N, 115°5′36″W, 1 August 1951, W. I. Findlay 228 (ACAD-30954); Coppermine [Kugluktuk], 67°51′N, 115°16′W, 2 July 1972, F. Fodor N 147 (UBC-V151905); Kugluktuk, 67.82205°N, 115.080317°W, 27 June 2006, J. Davis 609 (CAN-597652); Kugluktuk, rocky slopes of North Hill, 67°49′29.6″N, 115°6′31″W ± 50 m, 50 m, 29 June 2014, Saarela, Sokoloff & Bull 3047 (CAN, UBC); old riverbed of Fockler Creek, ca. 2.3 km SSE of Sandstone Rapids, Coppermine River, 67°25′45.7″N, 115°37′21.8″W ± 25 m, 166 m, 2 July 2014, Saarela, Sokoloff & Bull 3171 (ALA, CAN); Kugluk (Bloody Falls) Territorial Park, day-use area above Bloody Falls (at outhouse and fire pit), 67°44′36.8″N, 115°22′11.1″W ± 25 m, 28 m, 12 July 2014, Saarela, Sokoloff & Bull 3834 (ALTA, CAN, O); W of Kugluktuk on tundra flats above Coppermine River, S of 1 Coronation Drive and N of power plant, 67°49′28.97″N, 115°5′0.2″W ± 100 m, 8 m, 25 July 2014, Saarela, Sokoloff & Bull 4377 (CAN).

Vaccinium uliginosum L., Fig. S29B—Bilberry | Circumboreal–polar

Previously recorded from Kugluktuk (Cody, 1954b; Porsild & Cody, 1980). We made collections at Fockler Creek, Kugluktuk, Melville Creek and Kugluk (Bloody Falls) Territorial Park. Elsewhere in the Canadian Arctic recorded from Baffin, Banks, Coats, Cornwallis, Salisbury, Southampton and Victoria islands, and numerous mainland sites (Porsild & Cody, 1980; Cody, Scotter & Zoltai, 1989; Korol, 1992; Aiken et al., 2007; Saarela et al., 2013a). Elven et al. (2011) recognise five subspecies, taking into account earlier taxonomic work (e.g., Young, 1970) and the results of a molecular study (Alsos et al., 2005; also see Eidesen et al., 2007) that identified five plastid lineages in the circumpolar species. They reported two of these taxa for Canada, subsp. uliginosum and subsp. microphyllum (Lange) Tolm., but distinguishing morphological characters are unclear. For this reason we do not recognise infraspecific taxa. Vander Kloet (2009) did not recognise infraspecific taxa. Hultén (1968) distinguished these taxa on the basis of leaf size and growth habit: leaves <1 cm and plants low-growing in subsp. microphyllum vs. leaves larger and plants more upright. By these characteristics, our collection nos. 3187, 3524 and 3828 would be subsp. uliginosum and no. 3045 would be subsp. microphyllum.

Specimens Examined: Canada. Nunavut: Kitikmeot Region: Coppermine [Kugluktuk], 67°49′36″N, 115°5′36″W, 22 July 1951, W. I. Findlay 169 (ACAD-30924, DAO1178342 01-01000619900, UBC-V40755) Hultén (1968), 11 July 1951, W. I. Findlay 113 (DAO-178363 01-01000619899); Coppermine [Kugluktuk], 67°51′N, 115°16′W, 2 July 1972, F. Fodor N 149 (UBC-V151906); Kugluktuk, 67.81875°N, 115.0885°W, 10 July 2006, J. Davis 619 (CAN-597656); Kugluktuk, rocky slopes of North Hill, 67°49′29.6″N, 115°6′31″W ± 50 m, 50 m, 29 June 2014, Saarela, Sokoloff & Bull 3045 (ALA, ALTA, CAN); old riverbed of Fockler Creek, ca. 2.3 km SSE of Sandstone Rapids, Coppermine River, 67°25′45.7″N, 115°37′21.8″W ± 25 m, 166 m, 2 July 2014, Saarela, Sokoloff & Bull 3187 (CAN, MO, O); confluence of Coppermine River and Melville Creek, just W of Coppermine Mountains, 67°15′52″N, 115°30′55.3″W ± 350 m, 178–190 m, 7 July 2014, Saarela, Sokoloff & Bull 3524 (CAN, UBC); Kugluk (Bloody Falls) Territorial Park, rocky cliffs and ledges directly above (W side) of Bloody Falls, just S of heavily used day-use/fishing area, 67°44′40.1″N, 115°22′4.9″W ± 20 m, 8 m, 12 July 2014, Saarela, Sokoloff & Bull 3828 (CAN, MT, US).

Vaccinium vitis-idaea subsp. minus (Lodd.) Hultén—Mountain cranberry | Circumboreal–polar

Previously recorded from Kugluktuk (Cody, 1954b; Porsild & Cody, 1980). Elsewhere in the Canadian Arctic recorded from Baffin, Banks, Coats, Salisbury, Southampton and Victoria islands, and numerous mainland sites (Porsild & Cody, 1980; Korol, 1992; Cody & Reading, 2005; Aiken et al., 2007; Saarela et al., 2013a). Elven et al. (2011) recognised two subspecies, of which only subsp. minus occurs in North America. Porsild & Cody (1980) recognised the taxon as var. minus Lodd. Vander Kloet (2009) did not comment on, or recognise, infraspecific taxa.

Specimens Examined: Canada. Nunavut: Kitikmeot Region: Coppermine [Kugluktuk], 67°49′36″N, 115°5′36″W, 17 July 1951, W. I. Findlay 127 (DAO-170408 01-01000619896); Kugluktuk, 67.81875°N, 115.0885°W, 10 July 2006, J. Davis 622 (CAN-597651); old riverbed of Fockler Creek, ca. 2.3 km SSE of Sandstone Rapids, Coppermine River, 67°25′45.7″N, 115°37′21.8″W ± 25 m, 166 m, 2 July 2014, Saarela, Sokoloff & Bull 3185 (CAN, UBC); NW-facing slope just upstream of small tributary from its confluence with Fockler Creek, ca. 2.4 km SSW of Sandstone Rapids, Coppermine River, 67°25′46″N, 115°38′49.4″W ± 200 m, 149 m, 3 July 2014, Saarela, Sokoloff & Bull 3312 (ALA, CAN); confluence of Coppermine River and Melville Creek, just W of Coppermine Mountains, 67°15′52″N, 115°30′55.3″W ± 350 m, 178–190 m, 7 July 2014, Saarela, Sokoloff & Bull 3521 (ALTA, CAN); Kugluk (Bloody Falls) Territorial Park, day-use area above Bloody Falls (at outhouse and fire pit), 67°44′36.8″N, 115°22′11.1″W ± 25 m, 28 m, 12 July 2014, Saarela, Sokoloff & Bull 3835 (CAN, O).

Fabaceae [5/11]

Astragalus alpinus L., Fig. S30A—Alpine milk-vetch | Circumpolar–alpine

Previously recorded from Kugluktuk (Cody, 1954b; Porsild & Cody, 1980). We made collections at Kugluktuk, Fockler Creek, Coppermine Mountains and Kugluk (Bloody Falls) Territorial Park. Elsewhere in the Canadian Arctic recorded from Baffin, Banks, Coats, King William, Melville, Prince Patrick, Southampton and Victoria islands, and numerous mainland sites (Porsild & Cody, 1980; Cody, Scotter & Zoltai, 1989; Korol, 1992; Cody, Reading & Line, 2003; Cody & Reading, 2005; Aiken et al., 2007; Saarela et al., 2013a; Blondeau, 2015b). Taxonomy follows Porsild & Cody (1980, no infraspecific taxon indicated), Welsh (2007, as var. alpinus) and Elven et al. (2011). The latter authors recognise a second Arctic infraspecific taxon, the American Beringian subsp. alaskanus Hultén, which was treated as a synonym of var. alpinus by Welsh (2007).

Specimens Examined: Canada. Nunavut: Kitikmeot Region: Coppermine [Kugluktuk], 67°49′36″N, 115°5′36″W, 21 July 1951, W. I. Findlay 163 (DAO-183557 01-01000619625) Welsh (2007), 20 June 1951, W. I. Findlay 18 (DAO-183559 01-01000619624) Welsh (2007), 4 July 1951, W. I. Findlay 92 (DAO-183560 01-01000619623) Welsh (2007), 2 July 1951, W. I. Findlay 71 (DAO-183558 01-01000679621); Coppermine River, Fort Hearne–Bloody Falls [67.7761972°N, 115.2037222°W ± 7.5 km], 1931, A. M. Berry 13 (CAN-76421); Kugluktuk, along edge of Inuit St., just NW of Northern Store, 67°49′39.4″N, 115°5′41.3″W ± 5 m, 5 m, 30 June 2014, Saarela, Sokoloff & Bull 3102 (CAN, UBC); old riverbed of Fockler Creek, ca. 2.3 km SSE of Sandstone Rapids, Coppermine River, 67°25′45.7″N, 115°37′21.8″W ± 25 m, 166 m, 1 July 2014, Saarela, Sokoloff & Bull 3159 (CAN); slopes on E side of Coppermine River, N of its confluence with Fockler Creek, ca. 0.8 km SW of Sandstone Rapids, 67°26′36.9″N, 115°38′50.1″W ± 50 m, 128 m, 4 July 2014, Saarela, Sokoloff & Bull 3378 (CAN, US); confluence of Coppermine River and Melville Creek, just W of Coppermine Mountains, 67°15′52″N, 115°30′55.3″W ± 350 m, 178–190 m, 7 July 2014, Saarela, Sokoloff & Bull 3503 (CAN, NY, O, QFA); S-facing sandstone cliffs above Coppermine River, ca. 7.8 km NNE of Sandstone Rapids, 67°31′15.1″N, 115°36′19.1″W ± 50 m, 106 m, 8 July 2014, Saarela, Sokoloff & Bull 3656 (ALA, CAN); Kugluk (Bloody Falls) Territorial Park, rocky cliffs and ledges directly above (W side) of Bloody Falls, just S of heavily used day-use/fishing area, 67°44′40.1″N, 115°22′4.9″W ± 20 m, 8 m, 12 July 2014, Saarela, Sokoloff & Bull 3796 (ALTA, CAN); grassy sandy flats on extensive sandy floodplain of Coppermine River, below steep cliff above river and S of Kugluktuk, 67°48′54.3″N, 115°6′9.1″W ± 20 m, 5 m, 26 July 2014, Saarela, Sokoloff & Bull 4412 (CAN).

Astragalus richardsonii E. Sheld., Fig. S30B—Richardson’s milk-vetch | North American (NW)

Previously recorded from Kugluktuk (Cody, 1954b, as Astragalus aboriginum Richardson; Porsild & Cody, 1980). We made collections at Fockler Creek, Melville Creek, Coppermine Mountains, Kugluk (Bloody Falls) Territorial Park and Kugluktuk. This species has a fairly restricted distribution, occurring primarily on Banks and Victoria islands and adjacent Nunavut and Northwest Territories (Porsild & Cody, 1980; Aiken et al., 2007; Saarela et al., 2013a). Taxonomy follows Elven et al. (2011) and Porsild & Cody (1980). Barneby (1964) treated the taxon as the “Mackenzie variant” of Astragalus aboriginum, now recognised as a more southern and mostly non-Arctic taxon (Porsild & Cody, 1980; Elven et al., 2011). Welsh (2007) treated it as a synonym of Astragalus australis var. glabriusculus (Hook.) Isely, part of a large and variable Astragalus australis (L.) Lam. species complex. Aiken et al. (2007) treated it as Astragalus australis.

Specimens Examined: Canada. Nunavut: Kitikmeot Region: Coppermine River, Fort Hearne–Bloody Falls [67.7761972°N, 115.2037222°W ± 7.5 km], 1931, A. M. Berry 14 (CAN-76208); Coppermine [Kugluktuk], 67°49′36″N, 115°5′36″W, 5 July 1951, W. I. Findlay 93 (DAO-183602 01-01000619670), 27 June 1951, W. I. Findlay 41 (DAO-183601 01-01000619671); Coppermine [Kugluktuk], west of settlement, 6 July 1958, R. D. Wood s.n. (CAN-265503); NW-facing slope just upstream of small tributary from its confluence with Fockler Creek, ca. 2.4 km SSW of Sandstone Rapids, Coppermine River, 67°25′46″N, 115°38′49.4″W ± 200 m, 149 m, 3 July 2014, Saarela, Sokoloff & Bull 3314 (CAN, UBC); N side of Fockler Creek, ca. 1.9 km S of Sandstone Rapids, Coppermine River, 67°25′57.89″N, 115°38′3.9″W ± 10 m, 162 m, 4 July 2014, Saarela, Sokoloff & Bull 3330 (CAN); meadow just S of Tundra Lake, ca. 4.2 km SE of Sandstone Rapids, Coppermine River, 67°25′34.8″N, 115°33′27.8″W ± 20 m, 265 m, 5 July 2014, Saarela, Sokoloff & Bull 3435 (CAN); confluence of Coppermine River and Melville Creek, just W of Coppermine Mountains, 67°15′52″N, 115°30′55.3″W ± 350 m, 178–190 m, 7 July 2014, Saarela, Sokoloff & Bull 3537 (ALA, CAN); flats atop and upper slopes of Coppermine Mountains, N/W side of Coppermine River, 67°14′49.9″N, 115°38′43.7″W ± 200 m, 467 m, 9 July 2014, Saarela, Sokoloff & Bull 3763 (ALTA, CAN); Kugluk (Bloody Falls) Territorial Park, flats on top of mountain on W side of Coppermine River, just S of the start of Bloody Falls Rapids, 67°44′2.8″N, 115°23′39.3″W ± 250 m, 110 m, 14 July 2014, Saarela, Sokoloff & Bull 3999 (CAN); S edge of Kugluktuk, 67°49′7.9″N, 115°6′4.9″W ± 10 m, 42 m, 24 July 2014, Saarela, Sokoloff & Bull 4361 (CAN).

Hedysarum americanum (Michx. ex Pursh) Britton, Fig. 55—Alpine sweet-vetch, licorice root | Amphi-Beringian (E)–North American

Figure 55 Hedysarum americanum.

(A) Inflorescence, Kugluk (Bloody Falls) Territorial Park, Nunavut, 13 July 2014. (B) Habitat, Kugluk (Bloody Falls) Territorial Park, Nunavut, 17 July 2014. (C) Habit, Kugluk (Bloody Falls) Territorial Park, Nunavut, 19 July 2014. Photographs by R. D. Bull (A, C) and P. C. Sokoloff (B).

Previously recorded from Bloody Falls and Kugluktuk (Cody, 1954b; Porsild & Cody, 1980). We made collections at Fockler Creek, Melville Creek and Kugluk (Bloody Falls) Territorial Park. Elsewhere in the Canadian Arctic recorded from Banks and Victoria islands and the adjacent mainland extending east to the Wager Bay and Rankin Inlet areas (Porsild & Cody, 1980; Korol, 1992; Aiken et al., 2007; Saarela et al., 2013a; Bennett, 2015). It has also been treated as Hedysarum alpinum var. americanum Michx. ex Pursh (Porsild & Cody, 1980) and Hedysarum alpinum L. (Brouillet et al., 2010+). Elven et al. (2011) consider Hedysarum americanum and Hedysarum alpinum L. to be distinct species, the latter restricted to non-Arctic parts of Siberia; we follow their treatment.

Specimens Examined: Canada. Nunavut: Kitikmeot Region: Bloody Falls, 67°44′N, 115°23′W, 18 July 1951, W. I. Findlay 145 (ACAD-30955, DAO-26311 01-01000619944); Coppermine [Kugluktuk], 67°49′36″N, 115°5′36″W, 31 July 1951, W. I. Findlay 216 (DAO-26312 01-01000619945); Coppermine [Kugluktuk], vic. of hamlet and airstrip, 67.78°N, 115.5°W ± 3,615 m, 23 June 1999, C. L. Parker & I. Jonsdottir 9103 (ALA); Kugluktuk, upper tundra slope overlooking Coppermine River [67°49′36″N, 115°5′36″W], 16 July 2000, L. K. Benjamin s.n. (ACAD-ECS015862); Kugluktuk, upper tundra slope overlooking Coppermine River [67°49′36″N, 115°5′36″W], 16 July 2000, L. K. Benjamin s.n. (ACAD-ECS015863); Kugluktuk, 67.820883°N, 115.087233°W, 22 July 2006, J. Davis 637 (CAN-597641); old riverbed of Fockler Creek, ca. 2.3 km SSE of Sandstone Rapids, Coppermine River, 67°25′45.7″N, 115°37′21.8″W ± 25 m, 166 m, 2 July 2014, Saarela, Sokoloff & Bull 3173 (CAN, UBC); confluence of Coppermine River and Melville Creek, just W of Coppermine Mountains, 67°15′52″N, 115°30′55.3″W ± 350 m, 178–190 m, 7 July 2014, Saarela, Sokoloff & Bull 3532 (ALA, CAN); Kugluk (Bloody Falls) Territorial Park, rocky cliffs and ledges directly above (W side) of Bloody Falls, just S of heavily used day-use/fishing area, 67°44′40.1″N, 115°22′4.9″W ± 20 m, 8 m, 12 July 2014, Saarela, Sokoloff & Bull 3821 (ALTA, CAN, O).

Hedysarum boreale subsp. mackenziei (Richardson) Welsh, Fig. 56—Mackenzie’s sweet-vetch | Amphi-Beringian (E)–North American

Figure 56 Hedysarum boreale subsp. mackenziei.

(A) Habitat, Kugluk (Bloody Falls) Territorial Park, Nunavut, 19 July 2014. (B) Inflorescence, Saarela et al. 3377. (C) Habit, Saarela et al. 3377. Photographs by R. D. Bull (A, C) and P. C. Sokoloff (B).

Previously recorded from Bloody Falls and the mouth of the Rae River (Cody, 1954b; Porsild & Cody, 1980). We made collections at Fockler Creek, Big Creek and Kugluk (Bloody Falls) Territorial Park. Elsewhere in the Canadian Arctic recorded from Banks, Eglinton and Victoria islands, the adjacent mainland extending to the western Hudson Bay shore, and northern Quebec at a few sites barely extending into the Arctic ecozone (Porsild & Cody, 1980; Korol, 1992; Cody & Reading, 2005; Aiken et al., 2007; Saarela et al., 2013a; Blondeau, 2015b). This taxon has also been treated as Hedysarum mackenziei Richardson (Porsild & Cody, 1980; Elven et al., 2011), which was described from collections made along the Coppermine River between Point Lake, Northwest Territories and the Arctic coast (type material BM000793111, GH00032387, GH00032388, NY00005596) (Richardson, 1823). Cody (1954b) recognised a white-flowered form from Bloody Falls (Hedysarum mackenzii f. niveum B. Boivin). We did not see any white-flowered plants in 2014.

Specimens Examined: Canada. Nunavut: Kitikmeot Region: Bloody Falls, 67°44′N, 115°23′W, 18 July 1951, W. I. Findlay 144 (DAO-180346 01-01000619884); Rae River mouth, 2 August 1955, R. E. Miller 315 (CAN-241994); slopes on E side of Coppermine River, N of its confluence with Fockler Creek, ca. 0.8 km SW of Sandstone Rapids, 67°26′36.9″N, 115°38′50.1″W ± 50 m, 128 m, 4 July 2014, Saarela, Sokoloff & Bull 3377 (CAN, UBC); forest and slopes at confluence of Big Creek and Coppermine River, N side of Coppermine River, S side of Coppermine Mountains, 67°14′29.3″N, 116°2′44.5″W ± 250 m, 180–199 m, 7 July 2014, Saarela, Sokoloff & Bull 3540 (ALTA, CAN, MO, O); Kugluk (Bloody Falls) Territorial Park, rocky beach above Bloody Falls, W bank of Coppermine River, 67°44′18″N, 115°22′57.3″W ± 250 m, 34 m, 14 July 2014, Saarela, Sokoloff & Bull 3966 (CAN, MT, UBC, US); Kugluk (Bloody Falls) Territorial Park, W side of Coppermine River, just above Bloody Falls, 67°44′22.6″N, 115°22′52″W ± 20 m, 40 m, 16 July 2014, Saarela, Sokoloff & Bull 4110 (ALA, CAN).

Lathyrus japonicus Willd., Fig. 57—Beach pea | Amphi-Pacific/Beringian–North American–amphi-Atlantic–European

Figure 57 Lathyrus japonicus.

(A) Habitat, Saarela et al. 3713. (B) Habit, Saarela et al. 3713. Photographs by P. C. Sokoloff.

Previously recorded from Kugluktuk, as Lathyrus japonicus var. aleuticus (Greene) Fernald (Cody, 1954b; Porsild & Cody, 1980). We made collections along the beach west of Kugluktuk and on an island at the mouth of the Coppermine River. The species is at the edge of its range in the study area. Elsewhere on the mainland Canadian Arctic known from a few collections between the study area and Bathurst Inlet, the Mackenzie Delta area and a few sites along the eastern shore of Hudson Bay in northern Quebec (Porsild & Cody, 1980; Blondeau, 2015b). It is not recorded from the Canadian Arctic Archipelago. Taxonomy follows Elven et al. (2011).

Specimens Examined: Canada. Nunavut: Kitikmeot Region: Coppermine River, Fort Hearne–Bloody Falls [67.7761972°N, 115.2037222°W ± 7.5 km], 1931, A. M. Berry 15 (CAN-77945); Coppermine [Kugluktuk], 67°49′36″N, 115°5′36″W, 24 July 1981, W. I. Findlay 179 (DAO-180402 01-01000619940); Coppermine [Kugluktuk], 67°49′36″N, 115°5′36″W, 2 August 1995, T. Dolman 94 (LEA); unnamed island just E (ca. 3.3 km) of Kugluktuk at mouth of Coppermine River, 67°49′29.2″N, 115°1′3.2″W ± 50 m, 1 m, 8 July 2014, Saarela, Sokoloff & Bull 3713 (ALA, ALTA, CAN, UBC); sandy beach along Coronation Gulf, 3.9 km W of mouth of Coppermine River, 67°49′37.8″N, 115°10′31.8″W ± 50 m, 9 m, 23 July 2014, Saarela, Sokoloff & Bull 4336 (CAN).

Lupinus arcticus S. Watson subsp. arcticus, Fig. 58—Arctic lupine | North American (NW)

Figure 58 Lupinus arcticus.

(A) Inflorescence, Saarela et al. 3125. (B) Fruits, Kugluk (Bloody Falls) Territorial Park, Nunavut, 13 July 2014. Photographs by R. D. Bull.

Previously recorded from Kugluktuk (Cody, 1954b; Porsild & Cody, 1980). We made collections at Fockler Creek, Kugluk (Bloody Falls) Territorial Park and Kugluktuk. Elsewhere in the Canadian Arctic recorded from Victoria Island and scattered sites on adjacent mainland Nunavut and Northwest Territories (Porsild & Cody, 1980; Cody & Reading, 2005; Aiken et al., 2007; Saarela et al., 2013a). A second infraspecific taxon, subsp. subalpinus (Piper & B. L. Rob.) D. B. Dunn, occurs in non-Arctic parts of western North America (Brouillet et al., 2010+; Elven et al., 2011).

Specimens Examined: Canada. Nunavut: Kitikmeot Region: Coppermine [Kugluktuk], 67°49′36″N, 115°5′36″W, 20 June 1951, W. I. Findlay 19 (DAO-2039 01-01000619951) (Brouillet et al., 2010+; Elven et al., 2011), 3 July 1951, W. I. Findlay 75 (ACAD-30953, DAO-2040 01-01000619652); Coppermine River, Fort Hearne–Bloody Falls [67.7761972°N, 115.2037222°W ± 7.5 km], 1931, A. M. Berry 16 (CAN-75112); Coppermine [Kugluktuk], Coronation Gulf, at mouth of Coppermine River [67.822146°N, 115.078387°W ± 0.5 km], 4 August 1948, H. T. Shacklette 3337-a (CAN-200207); Kugluktuk, rocky outcrop, overlooking Coppermine River [67°49′36″N, 115°5′36″W], 21 June 2000, L. K. Benjamin s.n. (ACAD-ECS015875); Kugluktuk, rocky slopes of North Hill, 67°49′29.6″N, 115°6′31″W ± 50 m, 50 m, 29 June 2014, Saarela, Sokoloff & Bull 3054 (ALTA, CAN, O); flats on W side of Fockler Creek, above spruce forest in creek valley, ca. 2.2 km S of Sandstone Rapids, Coppermine River, 67°25′49″N, 115°37′55″W ± 50 m, 152 m, 1 July 2014, Saarela, Sokoloff & Bull 3125 (CAN, UBC); ridge N of Fockler Creek, ca. 2.1 km SSE of Sandstone Rapids, Coppermine River, 67°25′54″N, 115°37′30″W ± 25 m, 166 m, 2 July 2014, Saarela, Sokoloff & Bull 3206 (CAN); Kugluk (Bloody Falls) Territorial Park, flats above boardwalk W of Bloody Falls, 67°44′34.5″N, 115°22′27″W ± 100 m, 135 m, 13 July 2014, Saarela, Sokoloff & Bull 3922 (ALA, CAN).

Oxytropis arctica R. Br., Fig. S31A—Arctic oxytrope (locoweed) | Asian (N) & North American (NW)

Previously recorded from an island in the mouth of the Coppermine River (Cody, 1954b; Porsild & Cody, 1980). We made collections at Fockler Creek, Melville Creek and Coppermine Mountains. Elsewhere in the Canadian Arctic recorded from the western Arctic islands and the adjacent mainland (Porsild & Cody, 1980; Cody, Reading & Line, 2003; Aiken et al., 2007; Saarela et al., 2013a).

Specimens Examined: Canada. Nunavut: Kitikmeot Region: sandy island in Coppermine River at Coppermine [Kugluktuk], 67°50′N, 115°9′W, 30 June 1951, W. I. Findlay 56 (DAO-25890 01-01000619675, UBC-V196643); old riverbed of Fockler Creek, ca. 2.3 km SSE of Sandstone Rapids, Coppermine River, 67°25′45.7″N, 115°37′21.8″W ± 25 m, 166 m, 2 July 2014, Saarela, Sokoloff & Bull 3177 (CAN, UBC); confluence of Coppermine River and Melville Creek, just W of Coppermine Mountains, 67°15′52″N, 115°30′55.3″W ± 350 m, 178–190 m, 7 July 2014, Saarela, Sokoloff & Bull 3538 (ALA, ALTA, CAN); flats atop and upper slopes of Coppermine Mountains, N/W side of Coppermine River, 67°14′49.9″N, 115°38′43.7″W ± 200 m, 467 m, 9 July 2014, Saarela, Sokoloff & Bull 3764 (CAN).

Oxytropis arctobia Bunge, Fig. S31B—Blackish oxytrope (locoweed) | North American (N)

Several collections along the Arctic coast in the vicinity of the study area are mapped in Porsild & Cody (1980). Of these, we located one specimen from Rae River. The species is rare in the study area. We made collections at two small populations in Kugluk (Bloody Falls) Territorial Park and one on the east side of Bloody Falls outside the park. There is a recent collection from a nearby site to the east of the Coppermine River (66°44′0″N, 114°47′0″W, Reading 533, DAO-788054 01-01000619678). Elsewhere in the Canadian Arctic recorded from Baffin, Banks, Eglinton, King William, Melville, Southampton and Victoria islands, and some mainland Nunavut and Northwest Territories sites (Porsild & Cody, 1980; Cody & Reading, 2005; Aiken et al., 2007; Saarela et al., 2013a). The taxonomy of the Oxytropis nigrescens (Pall.) Fisch. ex DC. aggregate, which includes Oxytropis arctobia and Oxytropis bryophila (Greene) Jurtzev, is problematic, as discussed by Elven et al. (2011).

Specimens Examined: Canada. Nunavut: Kitikmeot Region: Rae River, west bank, 6 August 1955, R. E. Miller 317 (CAN-241996); Kugluk (Bloody Falls) Territorial Park, top of sandy ridge, ca. 0.75 km W of Bloody Falls, 67°44′45.7″N, 115°23′4.6″W ± 25 m, 56 m, 15 July 2014, Saarela, Sokoloff & Bull 4044 (CAN, UBC); Kugluk (Bloody Falls) Territorial Park, W side of Coppermine River, along ATV trail atop sand hill just below picnic bench/lookout area, 67°45′15.5″N, 115°22′31.9″W ± 3 m, 73 m, 17 July 2014, Saarela, Sokoloff & Bull 4150 (CAN); top of hill on E side of Bloody Falls, across Coppermine River from Kugluk (Bloody Falls) Territorial Park, 67°44′28.2″N, 115°22′0.3″W ± 50 m, 78 m, 19 July 2014, Saarela, Sokoloff & Bull 4178 (CAN).

Oxytropis deflexa subsp. foliolosa (Hook.) Cody, Fig. S32—Pendent pod oxytrope, locoweed | Amphi-Beringian–North American (W)

Previously recorded from Kugluktuk (Cody, 1954b; Porsild & Cody, 1980). We made collections at Fockler Creek and Kugluk (Bloody Falls) Territorial Park. This taxon has a scattered distribution in the Canadian Arctic, recorded from southern Baffin Island, Victoria Island, Tuktut Nogait National Park and vicinity, and mainland sites (Porsild & Cody, 1980; Cody & Reading, 2005; Aiken et al., 2007; Saarela et al., 2013a; L. J. Gillespie & J. M. Saarela, 2016, unpublished data; Blondeau, 2015b; Gillespie et al., 2015). This subspecies may be synonymous with subsp. dezhnevii (Jurtzev) Jurtzev from Far East Russia; if so the latter name has priority (Elven et al., 2011).

Specimens Examined: Canada. Nunavut: Kitikmeot Region: Coppermine [Kugluktuk], 67°49′36″N, 115°5′36″W, 21 July 1951, W. I. Findlay 160 (DAO-180161 01-01000619667); E side of Fockler Creek, just above its confluence with Coppermine River, ca. 1.1 km SW of Sandstone Rapids, 67°26′30.6″N, 115°39′4.3″W ± 50 m, 135 m, 4 4 July 2014, Saarela, Sokoloff & Bull 3366 (CAN, UBC); Kugluk (Bloody Falls) Territorial Park, rocky beach above Bloody Falls, W bank of Coppermine River, 67°44′18″N, 115°22′57.3″W ± 250 m, 34 m, 14 July 2014, Saarela, Sokoloff & Bull 3967 (ALA, CAN).

Oxytropis maydelliana Trautv., Fig. 59A—Maydell’ oxytrope (locoweed), Inuit carrot | Amphi-Beringian–North American (N)

Figure 59 Oxytropis maydelliana and Oxytropis varians.

Oxytropis maydelliana: (A) inflorescence, Saarela et al. 3154. Oxytropis varians: (B) inflorescence, Saarela et al. 4212. Photographs by R. D. Bull (A) and P. C. Sokoloff (B).

Previously recorded from Bloody Falls and Kugluktuk (Cody, 1954b; Porsild & Cody, 1980). We made collections at Fockler Creek, Melville Creek and Kugluk (Bloody Falls) Territorial Park. Distributed throughout the Canadian Arctic, excluding the high Arctic islands (Porsild & Cody, 1980; Cody, Scotter & Zoltai, 1989; Korol, 1992; Cody & Reading, 2005; Aiken et al., 2007; Saarela et al., 2013a; Blondeau, 2015b). Two subspecies (subsp. maydelliana and subsp. melanocephala (Hook.) A. E. Porsild) are recognised in Porsild & Cody (1980), but not in Elven et al. (2011), whom we follow.

Specimens Examined: Canada. Nunavut: Kitikmeot Region: Bloody Falls, 67°44′N, 115°23′W, 18 July 1951, W. I. Findlay 147 (DAO-18227); Coppermine [Kugluktuk], 67°49′36″N, 115°5′36″W, 10 July 1951, W. I. Findlay 106 (DAO-180242 01-01000619942) Elven et al. (2011), 30 July 1951, W. I. Findlay 211 (DAO-180244 01-01000619941, UBC-V196642); Kugluktuk, well drained upper tundra slope overlooking Coppermine River [67°49′36″N, 115°5′36″W], 16 July 2000, L. K. Benjamin s.n. (ACAD-ECS015864); old riverbed of Fockler Creek, ca. 2.3 km SSE of Sandstone Rapids, Coppermine River, 67°25′45.7″N, 115°37′21.8″W ± 25 m, 166 m, 1 July 2014, Saarela, Sokoloff & Bull 3154 (CAN, UBC); confluence of Coppermine River and Melville Creek, just W of Coppermine Mountains, 67°15′52″N, 115°30′55.3″W ± 350 m, 178–190 m, 7 July 2014, Saarela, Sokoloff & Bull 3527 (ALA, CAN); Kugluk (Bloody Falls) Territorial Park, flats above boardwalk W of Bloody Falls, 67°44′34.5″N, 115°22′27″W ± 100 m, 135 m, 13 July 2014, Saarela, Sokoloff & Bull 3920 (CAN); NW-facing moist to wet sedge meadow, drainage running into Coppermine River above Bloody Falls on SE side, 67°44′26.2″N, 115°22′11.8″W ± 15 m, 47 m, 19 July 2014, Saarela, Sokoloff & Bull 4194 (CAN).

Oxytropis varians (Rydb.) K. Schum., Fig. 59B—Late yellow locoweed | North American

The current concept of Oxytropis varians (Aiken et al., 2007) includes the more northern Oxytropis hyperborea A. E. Porsild, which has been reported from the study area (Porsild, 1943; Porsild & Cody, 1980), although we were unable to locate a voucher specimen for confirmation. We made collections at Fockler Creek, Big Creek and Bloody Falls. Elsewhere in the Canadian Arctic recorded from Banks and Victoria islands, Tuktut Nogait National Park and vicinity, and near Tuktoyaktuk (Porsild & Cody, 1980; Aiken et al., 2007; Saarela et al., 2013a). The species is at the eastern edge of its range in the study area. Welsh (1991) treated Oxytropis varians (as a variety of Oxytropis campestris (L.) DC.) and Oxytropis hyperborea as distinct taxa. The name Oxytropis hyperborea is apparently not treated in Elven et al. (2011).

Specimens Examined: Canada. Nunavut: Kitikmeot Region: second ridge N of Fockler Creek, ca. 1.9 km SSE of Sandstone Rapids, Coppermine River, 67°26′2.4″N, 115°37′26.5″W ± 25 m, 187 m, 2 July 2014, Saarela, Sokoloff & Bull 3209 (CAN, UBC); N side of Fockler Creek, ca. 1.9 km S of Sandstone Rapids, Coppermine River, 67°25′57.89″N, 115°38′3.9″W ± 10 m, 162 m, 4 4 July 2014, Saarela, Sokoloff & Bull 3329 (ALA, CAN); forest and slopes at confluence of Big Creek and Coppermine River, N side of Coppermine River, S side of Coppermine Mountains, 67°14′29.3″N, 116°2′44.5″W ± 250 m, 180–199 m, 7 July 2014, Saarela, Sokoloff & Bull 3557 (ALTA, CAN); SSW-facing slopes above start of Bloody Falls, SE side of Coppermine River, 67°44′12.5″N, 115°22′31″W ± 50 m, 50–60 m, 19 July 2014, Saarela, Sokoloff & Bull 4212 (CAN).

Gentianaceae [4/4]

The generic classification follows recent treatments of Gentianaceae (Struwe et al., 2002; Elven et al., 2011; Pringle, in press).

Comastoma tenellum (Rottb.) Toyok.—Slender gentian | Circumpolar-alpine

Previously recorded from Kugluktuk, as Gentiana tenella Rottb. (Cody, 1954b; Porsild & Cody, 1980), but we were unable to locate the voucher specimen (Findlay 242, DAO) to confirm. This is apparently a rare species throughout its range (Gillett, 1963; Porsild & Cody, 1980). Taxonomy follows Pringle (in press), who does not recognise infraspecific taxa as in Gillett (1957). Our single collection was gathered in Kugluktuk, where we encountered a substantial population in disturbed areas along a roadside and between buildings, associated with Artemisia tilesii, Hordeum jubatum, Poa pratensis subsp. alpigena and Salix glauca. Elsewhere in the Canadian Arctic known only from Banks and Southampton islands, northern Quebec and a few sites on mainland Nunavut (Porsild & Cody, 1980; Aiken et al., 2007; Saarela et al., 2013a).

Specimens Examined: Canada. Nunavut: Kitikmeot Region: Kugluktuk, roadside and flats between buildings, 67°49′27.4″N, 115°5′26.2″W ± 25 m, 29 m, 26 July 2014, Saarela, Sokoloff & Bull 4397 (CAN, UBC).

Gentianella propinqua (Richardson) J. M. Gillett subsp. propinqua, Fig. 60A—Small-flowered gentian, four-parted gentian | Amphi-Beringian (E)–North American (N)

Figure 60 Gentianella propinqua subsp. propinqua and Pinguicula villosa.

Gentianella propinqua subsp. propinqua: (A) habit, Saarela et al. 4188. Pinguicula villosa: (B) habit, Saarela et al. 4278. Photographs by R. D. Bull.

Previously recorded from Kugluktuk (Cody, 1954b; Porsild & Cody, 1980), but we were unable to locate the voucher (Findlay 240) reported in Cody (1954b) as Gentiana propinqua Richardson. We made collections at Fockler Creek, Kendall River, Bigtree River, Kugluk (Bloody Falls) Territorial Park and Kugluktuk. Elsewhere in the Canadian Arctic recorded from Banks and Victoria islands and the adjacent mainland, to approximately as far east as Bathurst Inlet (Porsild & Cody, 1980; Cody, Reading & Line, 2003; Aiken et al., 2007; Saarela et al., 2013a). Taxonomy follows Pringle (in press) who recognises the widespread nominate subspecies and subsp. aleutica (Cham. & Schltdl.) J. M. Gillett, endemic to the Aleutian Islands and mainland Alaska. Elven et al. (2011), with some hesitation, recognised two sympatric infraspecific taxa, subsp. propinqua and subsp. arctophila (Griseb.) Tzvelev, the latter recognised at species level (as Gentiana arctophila Griseb., distinct from Gentiana propinqua Richardson) in Porsild & Cody (1980) and treated as a synonym of subsp. propinqua by Pringle (in press). The subspecies propinqua is defined by having corolla lobes with bristle-tips and subsp. arctophila by having corolla lobes without bristle tips (Porsild & Cody, 1980). The two forms are often found in the same population (Pringle, in press), as was the case for our collection no. 4396. All our other collections comprise plants with bristle-tipped corolla lobes.

Specimens Examined: Canada. Nunavut: Kitikmeot Region: slopes on E side of Coppermine River, N of its confluence with Fockler Creek, ca. 0.8 km SW of Sandstone Rapids, 67°26′36.9″N, 115°38′50.1″W ± 50 m, 128 m, 4 July 2014, Saarela, Sokoloff & Bull 3391 (CAN); S side of Fockler Creek, ca. 2.7 SE of Sandstone Rapids, Coppermine River, 67°25′38.2″N, 115°36′54.9″W ± 50 m, 128 m, 5 July 2014, Saarela, Sokoloff & Bull 3407 (CAN); confluence of Coppermine and Kendall rivers (NW side of Coppermine River, S side of Kendall River), 67°6′51.1″N, 116°8′18.3″W ± 150 m, 220 m, 7 July 2014, Saarela, Sokoloff & Bull 3573 (CAN, UBC); confluence of Coppermine and Bigtree rivers, 66°56′23.8″N, 116°21′3.2″W ± 100 m, 265 m, 7 July 2014, Saarela, Sokoloff & Bull 3695 (CAN); flats atop and upper slopes of Coppermine Mountains, NW side of Coppermine River, 67°14′53.6″N, 115°38′37.9″W ± 15 m, 401 m, 9 July 2014, Saarela, Sokoloff & Bull 3757 (CAN); Kugluk (Bloody Falls) Territorial Park, rocky valley immediately SW of Bloody Falls, along rough marked section of Portage Trail, upper pond just W of Bloody Falls, 67°44′39.5″N, 115°22′28.9″W ± 10 m, 15 m, 13 July 2014, Saarela, Sokoloff & Bull 3898 (CAN); Kugluk (Bloody Falls) Territorial Park, SE-facing slope above small stream in deep gully that runs into Coppermine River just below Bloody Falls, ca. 1 km W of Bloody Falls, 67°44′41.2″N, 115°23′34.8″W ± 50 m, 49 m, 15 July 2014, Saarela, Sokoloff & Bull 4031 (CAN); SW-facing slopes of shallow gully in sand hills above Bloody Falls, SE side of Coppermine River across river from Kugluk (Bloody Falls) Territorial Park, 67°44′28.2″N, 115°22′3″W ± 15 m, 78 m, 19 July 2014, Saarela, Sokoloff & Bull 4188 (ALA, CAN); W of Kugluktuk on tundra flats above Coppermine River, S of 1 Coronation Drive and N of community power plant, 67°49′28.97″N, 115°5′0.2″W ± 100 m, 8 m, 22 July 2014, Saarela, Sokoloff & Bull 4254 (CAN); grassy vacant lot in Kugluktuk, 67°49′30.5″N, 115°5′29.3″W ± 15 m, 21 m, 24 July 2014, Saarela, Sokoloff & Bull 4343 (CAN); Kugluktuk, roadside and flats between buildings, 67°49′27.4″N, 115°5′26.2″W ± 25 m, 29 m, 26 July 2014, Saarela, Sokoloff & Bull 4396 (ALTA, CAN).

Gentianopsis detonsa (Rottb.) Ma subsp. detonsa—Sheared gentian | Amphi-Atlantic

Previously recorded from Kugluktuk and the mouth of the Rae River, as Gentianella detonsa Rottb. subsp. detonsa (Gillett, 1963), Gentiana detonsa (Cody, 1954b) and Gentiana richardsonii (Porsild & Cody, 1980). We were unable to locate the specimens from Kugluktuk (Findlay 240, 241, DAO). We collected this coastal taxon on an unnamed island at the mouth of the Coppermine River. It is not recorded from the Canadian Arctic Archipelago and is at the edge of its range in the study area. Taxonomy follows Pringle (in press) who recognises four subspecies, of which only the nominate one and subsp. yukonensis (J. M. Gillett) J. M. Gillett are recorded for the Arctic, the latter restricted to Alaska and Yukon. Gentiana richardsonii A. E. Porsild, recorded for the study area in Porsild & Cody (1980) and recognised by Elven et al. (2011), is treated as a synonym of this species by Pringle (in press).

Specimens Examined: Canada. Nunavut: Kitikmeot Region: Rae River, mouth, 1 August 1955, R. E. Miller 297 (CAN-242008); unnamed island just E (ca. 3.3 km) of Kugluktuk at mouth of Coppermine River, 67°49′29.2″N, 115°1′3.2″W ± 50 m, 1 m, 8 July 2014, Saarela, Sokoloff & Bull 3728 (CAN).

Lomatogonium rotatum (L.) Fr. subsp. rotatum—Marsh felwort | European (NE) & Asian (C-NE) & North American–amphi-Atlantic (W)

Previously recorded from the mouth of the Rae River (Porsild & Cody, 1980). We did not encounter this species in 2014. It is known from only a few collections elsewhere in the Canadian Arctic (Porsild & Cody, 1980; Cody, Reading & Line, 2003; Aiken et al., 2007; Saarela et al., 2013a). The other subspecies, subsp. tenuifolium (Griseb.) A. E. Porsild, is a western taxon that reaches the Arctic in Alaska and Yukon (Elven et al., 2011).

Specimens Examined: Canada. Nunavut: Rae River mouth [67.919444°N, 115.525°W], 1 August 1955, R. E. Miller 296 (CAN-242010).

Haloragaceae [1/1]

Myriophyllum sibiricum Kom.—Northern water milfoil | Circumboreal–polar | Noteworthy Record

Our collection, from Kugluk (Bloody Falls) Territorial Park, is the first for the study area and fills in a distribution gap between Victoria Island, the lower Brock River and the eastern shore of Great Slave Lake (Porsild & Cody, 1980; Saarela et al., 2013a). Elsewhere in the Canadian Arctic known from southern Baffin Island, Southampton Island, extreme southeast mainland Nunavut, Tuktut Nogait National Park and vicinity, coastal areas of western Northwest Territories and northern Quebec (Porsild & Cody, 1980; Aiken et al., 2007; Saarela et al., 2013a; Garneau, 2015b). Although our material is vegetative and identification is difficult, the plants are tentatively referred to this species, which is the most northerly ranging one in North America. The plants were uncommon, submersed in one foot of water along the rocky edge of a small pond, growing with Utricularia vulgaris and Equisetum fluviatile.

Specimens Examined: Canada. Nunavut: Kitikmeot Region: Kugluk (Bloody Falls) Territorial Park, rocky valley immediately SW of Bloody Falls, along rough marked section of Portage Trail, 67°44′34″N, 115°22′16″W ± 50 m, 20 m, 18 July 2014, Saarela, Sokoloff & Bull 4170 (CAN).

Lentibulariaceae [2/4]

Pinguicula villosa L., Fig. 60B—Hairy butterwort | Circumpolar-alpine | Noteworthy Record

Newly recorded for the study area, closing a distribution gap between Great Bear Lake (Porsild, 1943; Porsild & Cody, 1980), Bathurst Inlet (Cody, Scotter & Zoltai, 1984) and Hood River (Gould & Walker, 1997). We made two collections. At a Subarctic site near Tundra Lake this tiny insectivorous species was scattered on hummocks of the moss Sphagnum magellanicum Brid. within a wet sedge meadow with Carex aquatilis subsp. stans. minor, Carex capillaris subsp. fuscidula, Carex chordorrhiza, Equisetum arvense subsp. alpestre, Pinguicula vulgaris and Triglochin maritima. In the Arctic portion of the study area, it was growing on similar mossy hummocks in wet tundra around small ponds, with Andromeda polifolia, Betula glandulosa, Carex membranacea and Rubus chamaemorus. It is not recorded for the Canadian Arctic Archipelago (Aiken et al., 2007), and the study area is the species’ known northern limit in the Central Arctic. Elsewhere in Nunavut recorded from Chesterfield Inlet, Baker Lake (Ohenoja 107, CAN-588143; Wood s.n., CAN-265461), Rankin Inlet (Brunton & McIntosh 10495, CAN-565030, 10512, CAN-565186), Bathurst Inlet (Cody, Scotter & Zoltai, 1984) and a few other eastern mainland sites (Gussow, 1933; Porsild, 1950b; Savile & Calder, 1952; Porsild & Cody, 1980). It also reaches the Arctic in the Mackenzie River Delta area (Porsild, 1943; Porsild & Cody, 1980). It is easily distinguished from Pinguicula vulgaris (Crow, 2014), which is much more common and widespread in the Arctic.

Specimens Examined: Canada. Nunavut: Kitikmeot Region: tundra below Tundra Lake and Fockler Creek, ca. 4.1 km SE of Sandstone Rapids, Coppermine River, 67°25′20.7″N, 115°34′17.2″W ± 25 m, 271 m, 5 July 2014, Saarela, Sokoloff & Bull 3426 (CAN); ca. 0.5 km SW of Heart Lake, SW of Kugluktuk, 7.5 km SW of mouth of Coppermine River, 67°47′52″N, 115°14′14.4″W ± 350 m, 66 m, 23 July 2014, Saarela, Sokoloff & Bull 4278 (CAN).

Pinguicula vulgaris L. subsp. vulgaris, Fig. S33—Common butterwort | Amphi-Pacific–North American–amphi-Atlantic–European

Previously recorded from Kugluktuk (Cody, 1954b; Porsild & Cody, 1980). We made collections at Fockler Creek, Coppermine Mountains, Kugluk (Bloody Falls) Territorial Park and Kugluktuk. Elsewhere in the Canadian Arctic known from Victoria Island, southern Baffin Island and some mainland sites (Porsild & Cody, 1980; Aiken et al., 2007; Saarela et al., 2013a; Gillespie et al., 2015). Arctic plants are subsp. vulgaris, and subsp. macroceras (Link) Calder & R. L. Taylor is a non-Arctic taxon distributed in western North America (Elven et al., 2011).

Specimens Examined: Canada. Nunavut: Kitikmeot Region: Coppermine [Kugluktuk], 67°49′36″N, 115°5′36″W, 31 July 1951, W. I. Findlay 214 (DAO-11141 01-000456648, QFA0274611); sedge meadow at S end of small lake, on flats NW of Fockler Creek, ca. 1.9 km SSE of Sandstone Rapids, Coppermine River, 67°26′1.8″N, 115°37′30.5″W ± 20 m, 170 m, 2 July 2014, Saarela, Sokoloff & Bull 3233 (CAN); forest and slopes at confluence of Big Creek and Coppermine River, N side of Coppermine River, S side of Coppermine Mountains, 67°14′29.3″N, 116°2′44.5″W ± 250 m, 180–199 m, 7 July 2014, Saarela, Sokoloff & Bull 3541 (CAN); Kugluk (Bloody Falls) Territorial Park, rocky valley immediately SW of Bloody Falls, along rough marked section of Portage Trail, 67°44′34″N, 115°22′16″W ± 50 m, 20 m, 13 July 2014, Saarela, Sokoloff & Bull 3867 (CAN); SW-facing slopes of shallow gully in sand hills above Bloody Falls, SE side of Coppermine River across river from Kugluk (Bloody Falls) Territorial Park, 67°44′28.2″N, 115°22′3″W ± 15 m, 78 m, 19 July 2014, Saarela, Sokoloff & Bull 4190 (CAN); W of Kugluktuk on tundra flats above Coppermine River, S of 1 Coronation Drive and N of community power plant, 67°49′28.97″N, 115°5′0.2″W ± 100 m, 8 m, 22 July 2014, Saarela, Sokoloff & Bull 4266 (CAN).

Utricularia intermedia Hayne—Flat-leaved bladderwort | Circumboreal | Noteworthy Record

Newly recorded for the study area. Our single collection from the Fockler Creek area represents a northeastern range extension from the nearest known sites along eastern Great Bear Lake. The species was found submerged in shallow water in a small pool along a creek, where it was uncommon. Elsewhere in Nunavut recorded from the southeastern mainland (Porsild & Cody, 1980; Cody, 1996a) and Akimiski Island (Blaney & Kotanen, 2001).

Specimens Examined: Canada. Nunavut: Kitikmeot Region: S of Fockler Creek, along small tributary that runs into Fockler Creek, ca. 2.3 km S of Sandstone Rapids, Coppermine River, 67°25′44.9″N, 115°38′25.9″W ± 100 m, 152 m, 3 July 2014, Saarela, Sokoloff & Bull 3275 (CAN, UBC).

Utricularia vulgaris L., Fig. 61—Common bladderwort | Circumboreal | Noteworthy Record

Figure 61 Utricularia vulgaris.

(A) Habit, Saarela et al. 3877. (B) Bladders, Saarela et al. 3877. Photographs by J. M. Saarela.

Newly recorded for the study area. Our single collection from Kugluk (Bloody Falls) Territorial Park represents a northeastern range extension from the nearest known sites along eastern Great Bear Lake. Elsewhere in Nunavut recorded from the southeastern mainland (Porsild & Cody, 1980) and Akimiski Island (Blaney & Kotanen, 2001). The species was found submerged in shallow water along rocky edges of a small still pond, where it was locally common in 1–1.5 feet of water, growing with Equisetum fluviatile. Our collection is likely referable to Utricularia vulgaris subsp. vulgaris, the more widely distributed taxon in North America, but the plants did not have flowers, which are needed to identify infraspecific taxa (Crow, 2014).

Specimens Examined: Canada. Nunavut: Kitikmeot Region: Kugluk (Bloody Falls) Territorial Park, rocky valley immediately SW of Bloody Falls, along rough marked section of Portage Trail, 67°44′34″N, 115°22′16″W ± 50 m, 20 m, 13 July 2014, Saarela, Sokoloff & Bull 3877 (CAN, UBC).

Linaceae [1/1]

Linum lewisii Pursh subsp. lewisii—Lewis’s flax | North American (W) | Noteworthy Record

Newly recorded for the study area, where the species was uncommon at a site along south-facing slopes of the Coppermine River, growing with Dasiphora fruticosa and Plantago canescens. Our collection fills a distribution gap between Tuktut Nogait National Park, eastern Great Bear Lake, and a site between Bathurst Inlet and the study area (Porsild & Cody, 1980; Saarela et al., 2013a). On mainland Nunavut recorded as far east as Bathurst Inlet (Porsild & Cody, 1980). In the Canadian Arctic Archipelago known only from the Minto Inlet area of Victoria Island (Aiken et al., 2007). The other infraspecific taxon, subsp. lepagei (B. Boivin) Mosquin, is endemic to shores and islands along the southern part of Hudson Bay and James Bay where it is restricted to limestone outcrops (Mosquin, 1971; Elven et al., 2011).

Specimens Examined: Canada. Nunavut: Kitikmeot Region: S-facing sandstone cliffs above Coppermine River, ca. 7.8 km NNE of Sandstone Rapids, 67°31′15.1″N, 115°36′19.1″W ± 50 m, 106 m, 8 July 2014, Saarela, Sokoloff & Bull 3634 (CAN).

Linnaeaceae [1/1]

Linnaea borealis subsp. americana (J. Forbes) Hultén, Fig. 62—Twinflower | North American | Noteworthy Record

Figure 62 Linnaea borealis subsp. americana.

(A) Habit, Saarela et al. 3646. (B) Habit, Saarela et al. 3646. Photographs by R. D. Bull.

First record for the study area and first Arctic record for Nunavut. We made collections at Big Creek, a site north of Sandstone Rapids, Coppermine Mountains and Kugluk (Bloody Falls) Territorial Park. This species was recently reported from the nearby Big Bend area of the Coppermine River (Reading 17, DAO; Cody, Reading & Line, 2003) and a site east of the confluence of the Coppermine and Kendall rivers (Cody & Reading, 2005). Our collections represent a minor northern range extension with respect to these. The next-nearest records are from eastern Great Bear Lake (Porsild & Cody, 1980). Two of the three subspecies recognised in Linnaea borealis, subsp. borealis and subsp. americana, reach the Arctic in North America (Hultén, 1968; Elven et al., 2011). The subspecies americana is distributed across most of North America and has previously been reported from mainland Nunavut in the Subarctic Nueltin Lake area (Porsild, 1950b; Porsild & Cody, 1980). It reaches the Arctic elsewhere in the Mackenzie Delta area and Alaska (Porsild & Cody, 1980). It is fairly common in the study area, often found in large populations on south-facing slopes usually under willow thickets. Common associates include Arnica angustifolia, Betula glandulosa, Dasiphora fruticosa, Juniperus communis subsp. depressa, Lupinus arcticus, Pyrola grandiflora, Salix glauca, Salix reticulata and Vaccinium uliginosum.

Linnaea L. was previously included in Caprifoliaceae, but is now recognised in its own family, Linnaeaceae (Backlund & Pyck, 1998). Based on the results of molecular analyses, expansion of the traditionally monotypic Linnaea was recently proposed to accommodate 17 species previously recognised in different genera (Christenhusz, 2013); none of these reach the Arctic. For an alternate taxonomic solution see Wang et al. (2015).

Specimens Examined: Canada. Nunavut: Kitikmeot Region: forest and slopes at confluence of Big Creek and Coppermine River, N side of Coppermine River, S side of Coppermine Mountains, 67°14′29.3″N, 116°2′44.5″W ± 250 m, 180–199 m, 7 July 2014, Saarela, Sokoloff & Bull 3565 (CAN, UBC); S-facing slopes above Coppermine River and below spruce forest, ca. 7.8 km NNE of Sandstone Rapids, 67°31′16.2″N, 115°36′52.1″W ± 200 m, 110 m, 8 July 2014, Saarela, Sokoloff & Bull 3646 (CAN, MO, MT); flats atop and upper slopes of Coppermine Mountains, N/W side of Coppermine River, 67°36′58.7″N, 115°29′18.3″W ± 99 m, 50 m, 8 July 2014, Saarela, Sokoloff & Bull 3781 (CAN); Kugluk (Bloody Falls) Territorial Park, S-facing cliff (gabbro sill) above start of Bloody Falls, W side of Coppermine River, W side of Portage Trail, 67°44′23.2″N, 115°22′54.5″W ± 50 m, 57 m, 16 July 2014, Saarela, Sokoloff & Bull 4071 (ALA, ALTA, CAN); SSW-facing slopes above start of Bloody Falls, SE side of Coppermine River, 67°44′12.5″N, 115°22′31″W ± 50 m, 50–60 m, 19 July 2014, Saarela, Sokoloff & Bull 4203 (CAN).

Onagraceae [2/6]

Chamerion angustifolium (L.) Holub subsp. angustifolium, Fig. 63—Fireweed | Circumboreal–polar

Figure 63 Chamerion angustifolium subsp. angustifolium.

(A) Inflorescence, vicinity of lower Coppermine River, Nunavut, 7 July 2014. (B) Habit, Saarela et al. 4081. Photographs by R. D. Bull.

Recorded previously for the study area (Mosquin, 1966; Porsild & Cody, 1980), but we were unable to locate a voucher specimen. We made collections at Fockler Creek, Kendall River, Kugluk (Bloody Falls) Territorial Park, Heart Lake and Kugluktuk. This common boreal-Subarctic species is at its known northern limit in the study area. It is not recorded in the western Canadian Arctic Archipelago, but known in the east from southern Baffin Island (Aiken et al., 2007). It was previously recognised as Epilobium angustifolium L. subsp. angustifolium (Porsild & Cody, 1980). It flowers much later than other species in the study area and was similarly reported to be a late-flowering species in the Nueltin Lake area (Porsild, 1950b). The other infraspecific taxon, subsp. circumvagum (Mosquin) Hoch, is reported for the Arctic in Canada, but its exact Arctic distribution is unclear (Elven et al., 2011).

Specimens Examined: Canada. Nunavut: Kitikmeot Region: spruce forest along Fockler Creek, ca. 2.3 km SSE of Sandstone Rapids, Coppermine River, 67°25′45.7″N, 115°37′21.8″W ± 25 m, 166 m, 2 July 2014, Saarela, Sokoloff & Bull 3204 (CAN); confluence of Coppermine and Kendall rivers (NW side of Coppermine River, S side of Kendall River), 67°6′51.1″N, 116°8′18.3″W ± 150 m, 220 m, 7 July 2014, Saarela, Sokoloff & Bull 3575 (CAN, UBC); Kugluk (Bloody Falls) Territorial Park, SE-facing slope above small stream in deep gully that runs into Coppermine River just below Bloody Falls, ca. 1 km W of Bloody Falls, 67°44′41.2″N, 115°23′34.8″W ± 50 m, 49 m, 15 July 2014, Saarela, Sokoloff & Bull 4038 (ALA, CAN); Kugluk (Bloody Falls) Territorial Park, S-facing cliff (gabbro sill) above start of Bloody Falls, W side of Coppermine River, W side of Portage Trail, 67°44′23.2″N, 115°22′54.5″W ± 50 m, 57 m, 16 July 2014, Saarela, Sokoloff & Bull 4081 (CAN, MO, MT); W of Kugluktuk on tundra flats above Coppermine River, S of 1 Coronation Drive and N of community power plant, 67°49′28.97″N, 115°5′0.2″W ± 100 m, 8 m, 22 July 2014, Saarela, Sokoloff & Bull 4256 (ALTA, CAN); N side of Heart Lake, below rocky cliff, SW of Kugluktuk, 5.64 km SW of mouth of Coppermine River, 67°48′33.8″N, 115°12′52.9″W ± 15 m, 39 m, 23 July 2014, Saarela, Sokoloff & Bull 4317 (CAN, O).

Chamerion latifolium (L.) Holub, Fig. S34—Arctic fireweed, dwarf fireweed, river beauty, broad-leaved willow-herb | Circumpolar–alpine

Recorded previously from Bloody Falls and Kugluktuk (Cody, 1954b; Porsild & Cody, 1980). We made collections of this common species at Fockler Creek, Melville Creek and Kugluk (Bloody Falls) Territorial Park. Widespread throughout the Canadian Arctic (Porsild & Cody, 1980; Cody, Scotter & Zoltai, 1989; Korol, 1992; Cody & Reading, 2005; Aiken et al., 2007; Saarela et al., 2013a). White-flowered forms were scattered amongst normal-flowered individuals in a population on the southern sandy shores of Heart Lake and at Bigtree River; we did not document this variation with our collections. Previously recognised as Epilobium latifolium L. (Porsild & Cody, 1980).

Specimens Examined: Canada. Nunavut: Kitikmeot Region: Bloody Falls, 67°44′N, 115°23′W, 27 July 1951, W. I. Findlay 202 (ACAD-30945, DAO-133011 01-01000619653) (Porsild & Cody, 1980), 18 July 1951, W. I. Findlay 153 (DAO-133012 01-01000619654); Coppermine [Kugluktuk], 67°49′36″N, 115°5′36″W, 20 July 1951, W. I. Findlay 156 (DAO-133038 01-01000619657); Coppermine [Kugluktuk], vic. of hamlet and airstrip, 67.78°N, 115.5°W ± 3,615 m, 23 June 1999, C. L. Parker & I. Jonsdottir 9100 (ALA, as Epilobium latifolium); Kugluktuk, rocky slope at base of rock outcrop [67°49′36″N, 115°5′36″W], 16 July 2000, L. K. Benjamin s.n. (ACAD-ECS015861); old riverbed of Fockler Creek, ca. 2.3 km SSE of Sandstone Rapids, Coppermine River, 67°25′45.7″N, 115°37′21.8″W ± 25 m, 166 m, 2 July 2014, Saarela, Sokoloff & Bull 3169 (CAN, NY, QFA); Coppermine River, confluence of Coppermine River and Melville Creek, just W of Coppermine Mountains, 67°15′52″N, 115°30′55.3″W ± 350 m, 178–190 m, 7 July 2014, Saarela, Sokoloff & Bull 3488 (CAN, US); Kugluk (Bloody Falls) Territorial Park, rocky beach above Bloody Falls, W bank of Coppermine River, 67°44′18″N, 115°22′57.3″W ± 250 m, 34 m, 14 July 2014, Saarela, Sokoloff & Bull 3961 (CAN, WIN); Kugluk (Bloody Falls) Territorial Park, W side of Coppermine River, just above Bloody Falls, 67°44′22.6″N, 115°22′52″W ± 20 m, 40 m, 16 July 2014, Saarela, Sokoloff & Bull 4109 (CAN).

Epilobium cf. anagallidifolium Lam.—Alpine willowherb | Asian (C-NE)–amphi-Pacific/Beringian & North American (NE)–amphi-Atlantic–European (N/C) | Noteworthy Record

Our collection from Kugluktuk, tentatively identified as this taxon, is the first record for Nunavut and a range extension from the nearest known site along western Great Bear Lake (Porsild & Cody, 1980). The plants grew in moist to wet cracks along rocky cliffs just above the Coppermine River. This species was reported in Aiken et al. (2007) as occurring on “Nunavut Islands…range in the Canadian Arctic Archipelago not yet recorded”, but no evidence that it actually occurs there is provided: a specimen previously identified as this species from that area is noted and no distribution is given. Widespread in eastern and western North America, with a large gap in distribution in Central Canada (Porsild & Cody, 1980). If confirmed, this would be the northernmost record in the Canadian Arctic.

Specimens Examined: Canada. Nunavut: Kitikmeot Region: W of Kugluktuk on tundra flats above Coppermine River, S of 1 Coronation Drive and N of community power plant, 67°49′28.97″N, 115°5′0.2″W ± 100 m, 8 m, 22 July 2014, Saarela, Sokoloff & Bull 4263 (CAN).

Epilobium arcticum Sam.—Arctic willowherb | Nearly circumpolar | Noteworthy Record

Newly recorded for the study area and a southern range extension. The species was uncommon in low shrub tundra in Kugluk (Bloody Falls) Territorial Park, growing with Betula glandulosa, Eriophorum vaginatum subsp. vaginatum, Rhododendron lapponicum, Rumex arcticus and Salix spp. Known from elsewhere on mainland Nunavut from a few collections from the Hudson Bay area, Boothia Peninsula and the Bissett Lake area (Porsild & Cody, 1980; Cody, Reading & Line, 2003). The nearest collections are from Cambridge Bay and Ulukhaktok on adjacent Victoria Island (Porsild & Cody, 1980; Aiken et al., 2007) and the species is widespread throughout the Canadian Arctic Archipelago (Aiken et al., 2007). It differs from Epilobium palustre by its remotely denticulate leaves (vs. entire), and from Epilobium davuricum by its narrowly oblong leaves (vs. linear).

Specimens Examined: Canada. Nunavut: Kitikmeot Region: Kugluk (Bloody Falls) Territorial Park, flats above boardwalk W of Bloody Falls, 67°44′34.5″N, 115°22′27″W ± 100 m, 135 m, 16 July 2014, Saarela, Sokoloff & Bull 4052 (CAN).

Epilobium davuricum Fisch. ex Horn.—Dahurian willowherb | Circumboreal-polar | Noteworthy Record

Our collection from Kugluk (Bloody Falls) Territorial Park is the first record for the study area, and represents a range extension from the nearest known collections from eastern Great Bear Lake (Porsild, 1943; Porsild & Cody, 1980). The study area is the known northern limit of the species in the Central Arctic. It is not recorded for the Canadian Arctic Archipelago (Aiken et al., 2007). Elsewhere in Nunavut recorded from the Kazan River area (Porsild, 1943; Porsild & Cody, 1980), Bissett and Bernier Lake areas (Cody, Reading & Line, 2003) and T-Bone Lake (Cody & Reading, 2005). Our specimen was growing in luxuriant grass/sedge vegetation along an ATV trail, with Arctagrostis latifolia subsp. latifolia, Bistorta vivipara, Juncus leucochlamys, Poa arctica subsp. arctica and Salix spp. It was taken as part of a mixed collection, of which 4102a is Epilobium palustre. Epilobium davuricum differs from Epilobium palustre by its remotely denticulate leaves (vs. entire), and from Epilobium arcticum by its linear leaves (vs. narrowly oblong).

Specimens Examined: Canada. Nunavut: Kitikmeot Region: Kugluk (Bloody Falls) Territorial Park, along wet, muddy, and deeply pitted ATV trail ca. 1 km W of Bloody Falls, 67°44′33.2″N, 115°23′30″W ± 20 m, 73 m, 16 July 2014, Saarela, Sokoloff & Bull 4102b (CAN).

Epilobium palustre L., Fig. 64A—Marsh willowherb | Circumboreal-polar

Figure 64 Epilobium palustre and Castilleja caudata.

Epilobium palustre: (A) habit, Saarela et al. 4103. Castilleja caudata: (B) habitat, Saarela et al. 3183. Photographs by P. C. Sokoloff (A) and J. M. Saarela (B).

Previously recorded from Kugluktuk (Cody, 1954b; Porsild & Cody, 1980). We made collections at Kugluk (Bloody Falls) Territorial Park, Heart Lake and Kugluktuk. This is the most common Epilobium species in the study area. Elsewhere in the Canadian Arctic recorded from Southampton Island and numerous mainland sites (Porsild & Cody, 1980; Korol, 1992; Cody, Reading & Line, 2003; Cody & Reading, 2005; Aiken et al., 2007).

Specimens Examined: Canada. Nunavut: Kitikmeot Region: Coppermine [Kugluktuk], 67°49′36″N, 115°5′36″W, 4 August 1951, W. I. Findlay 248 (DAO-134304 01-01000616663); Kugluk (Bloody Falls) Territorial Park, flats above Coppermine River valley, ca. 1 km W of Bloody Falls, 67°44′31.8″N, 115°24′25.6″W ± 5 m, 97 m, 16 July 2014, Saarela, Sokoloff & Bull 4103 (CAN, UBC); Kugluk (Bloody Falls) Territorial Park, along wet, muddy, and deeply pitted ATV trail ca. 1 km W of Bloody Falls, 67°44′33.2″N, 115°23′30″W ± 20 m, 73 m, 16 July 2014, Saarela, Sokoloff & Bull 4102a (CAN); N side of Heart Lake, below rocky cliff, SW of Kugluktuk, 5.64 km SW of mouth of Coppermine River, 67°48′33.8″N, 115°12′52.9″W ± 15 m, 39 m, 23 July 2014, Saarela, Sokoloff & Bull 4313 (ALTA, CAN, O); W of Kugluktuk on tundra flats above Coppermine River, S of 1 Coronation Drive and N of power plant, 67°49′28.97″N, 115°5′0.2″W ± 100 m, 8 m, 25 July 2014, Saarela, Sokoloff & Bull 4366 (ALA, CAN).

Orobanchaceae [2/11]

Castilleja caudata (Pennell) Rebrist., Fig. 64B—Pale paintbrush | Asian (NE)–amphi-Beringian | Noteworthy Record

Newly recorded for the study area. We made collections at Fockler Creek, Kugluk (Bloody Falls) Territorial Park and Kugluktuk, which close a distribution gap between the Bathurst Inlet area, Hood River, Great Bear Lake and Tuktut Nogait National Park and vicinity (Porsild & Cody, 1980; Gould & Walker, 1997; Saarela et al., 2013a). The study area represents its known northern limit in Nunavut. It is not recorded from the Canadian Arctic Archipelago (Aiken et al., 2007).

The Canadian Arctic Castilleja species are difficult to distinguish. Taxonomy here follows Porsild & Cody (1980) and Elven et al. (2011), and our identifications are based on keys and descriptions in Pennell (1934) and Porsild & Cody (1980), and study of type specimens and other material at CAN, much of which has been recently annotated by Castilleja authority M. Egger (WTU). Castilleja caudata is part of the widespread and taxonomically challenging Castilleja pallida (L.) Spreng. complex (Egger, 2008). Pennell (1934), in his taxonomic treatment of the group in Alaska and northwestern Canada, treated it as Castilleja pallida subsp. caudata Pennell, and Egger (2008) recognised it as a Castilleja pallida var. caudata (Pennell) B. Boivin. Two of our collections (nos. 4418 and 4419) are unusual in having branched stems; both were gathered on a sandy floodplain of the Coppermine River. No. 4418 was recorded as being uncommon in a large stand of “typical” (unbranched) Castilleja caudata. A specimen from Bloody Falls (Findlay 200) identified as Castilleja pallida subsp. mexiae Pennell var. elegans (Malte) B. Boivin (=Castilleja elegans) by W. J. Cody in 1952 (Cody, 1954b) and as Castilleja pallida subsp. pallida by Cody and A. E. Porsild in 1967, is placed here.

Specimens Examined: Canada. Nunavut: Kitikmeot Region: Coppermine [Kugluktuk], 67°49′36″N, 115°5′36″W, 27 July 1951, W. I. Findlay 200 (DAO-179175 01-01000620019); old riverbed of Fockler Creek, ca. 2.3 km SSE of Sandstone Rapids, Coppermine River, 67°25′45.7″N, 115°37′21.8″W ± 25 m, 166 m, 2 July 2014, Saarela, Sokoloff & Bull 3183 (CAN, UBC); S of Fockler Creek, along small tributary that runs into Fockler Creek, ca. 2.3 km S of Sandstone Rapids, Coppermine River, 67°25′44.9″N, 115°38′25.9″W ± 100 m, 152 m, 3 July 2014, Saarela, Sokoloff & Bull 3240 (CAN, US, WIN); Kugluk (Bloody Falls) Territorial Park, rocky cliffs and ledges directly above (W side) of Bloody Falls, just S of heavily used day-use/fishing area, 67°44′40.1″N, 115°22′4.9″W ± 20 m, 8 m, 12 July 2014, Saarela, Sokoloff & Bull 3819a (CAN, MT); Kugluk (Bloody Falls) Territorial Park, rocky beach above Bloody Falls, W bank of Coppermine River, 67°44′18″N, 115°22′57.3″W ± 250 m, 34 m, 14 July 2014, Saarela, Sokoloff & Bull 3977 (ALA, CAN); Kugluk (Bloody Falls) Territorial Park, S-facing cliff (gabbro sill) above start of Bloody Falls, W side of Coppermine River, W side of Portage Trail, 67°44′23.2″N, 115°22′54.5″W ± 50 m, 57 m, 16 July 2014, Saarela, Sokoloff & Bull 4073 (CAN); Kugluk (Bloody Falls) Territorial Park, S-facing cliff (gabbro sill) above start of Bloody Falls, W side of Coppermine River, W side of Portage Trail, 67°44′23.2″N, 115°22′54.5″W ± 50 m, 57 m, 16 July 2014, Saarela, Sokoloff & Bull 4074 (CAN); SW-facing slopes of shallow gully in sand hills above Bloody Falls, SE side of Coppermine River across river from Kugluk (Bloody Falls) Territorial Park, 67°44′28.2″N, 115°22′3″W ± 15 m, 78 m, 19 July 2014, Saarela, Sokoloff & Bull 4187 (ALTA, CAN); SW-facing slopes of shallow gully in sand hills above Bloody Falls, SE side of Coppermine River across river from Kugluk (Bloody Falls) Territorial Park, 67°44′28.2″N, 115°22′3″W ± 15 m, 78 m, 19 July 2014, Saarela, Sokoloff & Bull 4189 (CAN, O); grassy sandy flats on extensive sandy floodplain of Coppermine River, below steep cliff above river and S of Kugluktuk, 67°48′54.3″N, 115°6′9.1″W ± 20 m, 5 m, 26 July 2014, Saarela, Sokoloff & Bull 4418 (CAN); grassy sandy flats on extensive sandy floodplain of Coppermine River, below steep cliff above river and S of Kugluktuk, 67°48′54.3″N, 115°6′9.1″W ± 20 m, 5 m, 26 July 2014, Saarela, Sokoloff & Bull 4419 (CAN, O).

Castilleja elegans Malte, Fig. 65—Elegant paintbrush | Amphi-Beringian–North American (NW)

Figure 65 Castilleja elegans.

(A) Habit, vicinity of Kugluktuk, Nunavut, 26 July 2014. (B) Inflorescence, vicinity of Kugluktuk, Nunavut, 26 July 2014. Photographs by R. D. Bull.

Recorded previously from Kugluktuk (Cody, 1954b; Porsild & Cody, 1980). We made collections at Heart Lake and Kugluktuk. Elsewhere in the Canadian Arctic recorded from Banks and Victoria islands, and numerous sites on mainland Nunavut (Porsild & Cody, 1980; Aiken et al., 2007). In the course of examining material in CAN, where the holotype for the name Castilleja elegans is housed (Cox & O’Neill 433, CAN-96165), a previously unrecorded isotype was found (Cox & O’Neill 433, CAN-504545). Taxonomy follows numerous authors who have recognised this taxon at species level (Malte, 1934; Porsild & Cody, 1980; Egger, 2008; Elven et al., 2011). Pennell (1934) recognised it as Castilleja pallida subsp. elegans (Malte) Pennell.

Specimens Examined: Canada. Nunavut: Kitikmeot Region: Kugluktuk, 67.8272°N, 115.113617°W, 3 July 2006, J. Davis 615 (CAN-597638); Coppermine [Kugluktuk], vicinity of post [67°49′36″N, 115°5′36″W ± 1.5 km], 26 July 1949, A. E. Porsild 17184 (CAN-128080); Coppermine River, Fort Hearne–Bloody Falls [67.7761972°N, 115.2037222°W ± 7.5 km], 1931, A. M. Berry 21 (CAN-96166); Coppermine [Kugluktuk], rocky ledge overlooking Coppermine River [67.816375°N, 115.1002722°W ± 350 m], 11 July 1958, R. D. Wood s.n. (CAN-265479); Coppermine [Kugluktuk] [67°49′36″N, 115°5′36″W ± 1.5 km], 8 July 1955, R. E. Miller 51 (CAN-242011); Heart Lake, SW of Kugluktuk, 6.4 km SW of mouth of Coppermine River, 67°48′7.8″N, 115°13′22.7″W ± 350 m, 33 m, 23 July 2014, Saarela, Sokoloff & Bull 4298 (CAN); W of Kugluktuk on tundra flats above Coppermine River, S of 1 Coronation Drive and N of power plant, 67°49′28.97″N, 115°5′0.2″W ± 100 m, 8 m, 25 July 2014, Saarela, Sokoloff & Bull 4381 (CAN, UBC).

Castilleja raupii Pennell—Raup’s paintbrush | North American (N) | Noteworthy Record

Newly recorded for the study area. Our collections, from Melville Creek and Kugluk (Bloody Falls) Territorial Park, and one from Kugluktuk gathered in 2013, represent a range extension from the nearest known collection from Great Bear Lake (Porsild & Porsild 5184, CAN-96181, det. M. Egger 2000–2001). The study area represents the known northern limit of the species in Nunavut. Elsewhere in Nunavut known from a few other Arctic locations on the mainland (Porsild & Cody, 1980; specimens at CAN; Korol, 1992) and Akimiski Island (Blaney & Kotanen, 2001). One of our collections was taken as part of a mixed collection with Castilleja caudata (no. 3819a).

Specimens Examined: Canada. Nunavut: Kugluktuk, airport, 21 July 2013, 67.81749°N, 115.13449°W, B. A. Bennett 13-0332 (BABY, det. B. A. Bennett, July 2013); Kitikmeot Region: confluence of Coppermine River and Melville Creek, just W of Coppermine Mountains, 67°15′52″N, 115°30′55.3″W ± 350 m, 178–190 m, 7 July 2014, Saarela, Sokoloff & Bull 3502 (CAN, K, UBC, US); Kugluk (Bloody Falls) Territorial Park, rocky cliffs and ledges directly above (W side) of Bloody Falls, just S of heavily used day-use/fishing area, 67°44′40.1″N, 115°22′4.9″W ± 20 m, 8 m, 12 July 2014, Saarela, Sokoloff & Bull 3819b (CAN).

Pedicularis albolabiata (Hultén) Kozhevn., Fig. S35—Sudetic lousewort, white-lipped lousewort | Asian (N)–amphi-Beringian–North American (N)

Recorded previously from Bloody Falls and Kugluktuk, as Pedicularis sudetica (Cody, 1954b; Porsild & Cody, 1980). We made collections at Fockler Creek, Big Creek, Kendall River, Kugluk (Bloody Falls) Territorial Park and on an island in the mouth of the Coppermine River. Widespread throughout the Canadian Arctic Archipelago and the adjacent mainland Arctic (Porsild & Cody, 1980; Aiken et al., 2007). Pedicularis albolabiata and Pedicularis arctoeuropaea were previously treated as Pedicularis sudetica Willd., sometimes as subspecies. Molau & Murray (1996) revised the Arctic-alpine Pedicularis sudetica complex and recognised these taxa as species. A key to the group in Arctic Canada is given in Saarela et al. (2013a).

Specimens Examined: Canada. Nunavut: Kitikmeot Region: Bloody Falls, 67°44′N, 115°23′W, 18 July 1951, W. I. Findlay 152 (DAO-179476 01-01000619959); Coppermine [Kugluktuk], 67°49′36″N, 115°5′36″W, 24 July 1951, W. I. Findlay 174 (DAO-179506 01-01000619960); Coppermine [Kugluktuk], 67°49′36″N, 115°5′36″W, 3 July 1951, W. I. Findlay 80 (DAO-179477 01-01000619958); Kugluktuk, upper tundra slope overlooking Coppermine River [67°49′36″N, 115°5′36″W], 16 July 2000, L. K. Benjamin s.n. (ACAD-ECS015871); sedge meadow adjacent to small lake on flats N of Fockler Creek, ca. 1.5 km SSE of Sandstone Rapids, Coppermine River, 67°26′8.8″N, 115°37′35.9″W ± 20 m, 168 m, 2 July 2014, Saarela, Sokoloff & Bull 3230a (CAN, O); S of Fockler Creek, along small tributary that runs into Fockler Creek, ca. 2.3 km S of Sandstone Rapids, Coppermine River, 67°25′44.9″N, 115°38′25.9″W ± 100 m, 152 m, 3 July 2014, Saarela, Sokoloff & Bull 3252 (CAN, UBC); forest and slopes at confluence of Big Creek and Coppermine River, N side of Coppermine River, S side of Coppermine Mountains, 67°14′29.3″N, 116°2′44.5″W ± 250 m, 180–199 m, 7 July 2014, Saarela, Sokoloff & Bull 3556 (CAN); confluence of Coppermine and Kendall rivers (NW side of Coppermine River, S side of Kendall River), 67°6′51.1″N, 116°8′18.3″W ± 150 m, 220 m, 7 July 2014, Saarela, Sokoloff & Bull 3580 (ALA, CAN); unnamed island just E (ca. 3.3 km) of Kugluktuk at mouth of Coppermine River, 67°49′29.2″N, 115°1′3.2″W ± 50 m, 1 m, 8 July 2014, Saarela, Sokoloff & Bull 3719 (CAN, MO, MT); Kugluk (Bloody Falls) Territorial Park, flats above boardwalk W of Bloody Falls, 67°44′34.5″N, 115°22′27″W ± 100 m, 135 m, 13 July 2014, Saarela, Sokoloff & Bull 3917 (ALTA, CAN); Kugluk (Bloody Falls) Territorial Park, along Portage Trail at top of ridge on W bank of Coppermine River, near start of Bloody Falls rapids, 67°44′22.5″N, 115°22′40.6″W ± 10 m, 46 m, 14 July 2014, Saarela, Sokoloff & Bull 3930 (CAN).

Pedicularis arctoeuropaea (Hultén) Molau & D. F. Murray—Sudetic lousewort, arctoeuropean lousewort | European (N)–Asian (N)–amphi-Beringian | Noteworthy Record

First report of the species from the study area, based on the re-determination of a specimen (Wood s.n.) previously identified as Pedicularis sudetica (see above) and our new collection from Richardson Bay. At that site, the species was scattered along the upper edges of the estuary, growing with Carex subspathacea, Potentilla anserina, Puccinellia phryganodes and Stellaria humifusa. These close a distribution gap between the vicinity of the mouth of the Brock River (Saarela et al., 2013a) and an island in Coronation Gulf northeast of the study area (Aiken et al., 2007). Elsewhere in the Canadian Arctic recorded from a few sites on mainland Nunavut as well as Banks, Coats, Melville and Victoria islands (Aiken et al., 2007). All of these are further east than the distribution given in Molau & Murray (1996).

Specimens Examined: Canada. Nunavut: Kitikmeot Region: Coppermine [Kugluktuk], near school in settlement [67°49′36″N, 115°5′36″W ± 1.5 km], 12 July 1958, R. D. Wood s.n. (CAN-265464); Richardson Bay, confluence of Richardson and Rae rivers at Coronation Gulf, ca. 20 km WNW of Kugluktuk, 67°54′11.2″N, 115°32′27.4″W ± 200 m, 0 m, 8 July 2014, Saarela, Sokoloff & Bull 3676 (CAN).

Pedicularis capitata Adams, Fig. 66A—Capitate lousewort, bananas | Asian (N)–amphi-Beringian–North American (N)

Figure 66 Pedicularis capitata and Pedicularis flammea.

Pedicularis capitata: (A) habit, Saarela et al. 3924. Pedicularis flammea: (B) habit, Saarela et al. 3242. Photographs by R. D. Bull.

Recorded previously from Kugluktuk (Cody, 1954b; Porsild & Cody, 1980). We made collections at Fockler Creek, along the Coppermine River between Escape Rapids and Muskox River, and Kugluk (Bloody Falls) Territorial Park. Widespread throughout the Canadian Arctic Archipelago and the mainland Arctic (Porsild & Cody, 1980; Korol, 1992; Cody & Reading, 2005; Aiken et al., 2007).

Specimens Examined: Canada. Nunavut: Kitikmeot Region: Coppermine [Kugluktuk], 67°49′36″N, 115°5′36″W, 3 July 1951, W. I. Findlay 76 (DAO-179303 01-01000620142); Coppermine [Kugluktuk], 67°49′36″N, 115°5′36″W, 30 June 1951, W. I. Findlay 56 (DAO-179304 01-01000620143); Kugluktuk, upper tundra slope overlooking Coppermine River [67°49′36″N, 115°5′36″W], 16 July 2000, L. K. Benjamin s.n. (ACAD-ECS015874); old riverbed of Fockler Creek, ca. 2.3 km SSE of Sandstone Rapids, Coppermine River, 67°25′48″N, 115°37′33″W ± 25 m, 153 m, 1 July 2014, Saarela, Sokoloff & Bull 3145 (CAN); S-facing slopes on W side of Coppermine River, about halfway between Escape Rapids and Muskox Rapids, 67°31′18.2″N, 115°36′20.1″W ± 150 m, 115 m, 8 July 2014, Saarela, Sokoloff & Bull 3624 (CAN); Kugluk (Bloody Falls) Territorial Park, flats above boardwalk W of Bloody Falls, 67°44′34.5″N, 115°22′27″W ± 100 m, 135 m, 13 July 2014, Saarela, Sokoloff & Bull 3924 (CAN).

Pedicularis flammea L., Fig. 66B—Red-tipped lousewort | North American (N)–amphi-Atlantic (W) | Noteworthy Record

Newly recorded for the study area. Our collection closes a distribution gap between Bathurst Inlet, Tuktut Nogait National Park and vicinity and eastern Great Bear Lake (Porsild & Cody, 1980; Saarela et al., 2013a). The species was uncommon in a Subarctic mesic meadow along a creek near Fockler Creek growing with Arctous rubra, Betula glandulosa, Carex scirpoidea subsp. scirpoidea, Hedysarum americanum, Lupinus arcticus, Luzula confusa, Luzula nivalis and Vaccinium uliginosum. Elsewhere in the Canadian Arctic recorded from the eastern Arctic Archipelago, mainland Nunavut and northern Quebec (Porsild & Cody, 1980; Korol, 1992; Cody, Reading & Line, 2003; Aiken et al., 2007; Saarela et al., 2013a). In the western portion of its range not recorded from north of the mainland.

Specimens Examined: Canada. Nunavut: Kitikmeot Region: S of Fockler Creek, along small tributary that runs into Fockler Creek, ca. 2.3 km S of Sandstone Rapids, Coppermine River, 67°25′44.9″N, 115°38′25.9″W ± 100 m, 152 m, 3 July 2014, Saarela, Sokoloff & Bull 3242 (CAN).

Pedicularis labradorica Wirsing, Fig. S36A—Labrador lousewort | Asian (N/C)–amphi-Beringian–North American (N)

Recorded previously from Kugluktuk (Cody, 1954b; Porsild & Cody, 1980), but we were unable to locate the voucher specimen (Findlay 136) for confirmation. We made collections at Fockler Creek, Melville Creek and Kugluk (Bloody Falls) Territorial Park. Elsewhere in the Canadian Arctic recorded from Banks Island, southern Baffin Island and several sites on mainland Nunavut and Northwest Territories (Porsild & Cody, 1980; Cody, Scotter & Zoltai, 1989; Korol, 1992; Aiken et al., 2007; Saarela et al., 2013a; Bennett, 2015).

Specimens Examined: Canada. Nunavut: Kitikmeot Region: S of Fockler Creek, along small tributary that runs into Fockler Creek, ca. 2.3 km S of Sandstone Rapids, Coppermine River, 67°25′44.9″N, 115°38′25.9″W ± 100 m, 152 m, 3 July 2014, Saarela, Sokoloff & Bull 3239 (CAN); S side of Fockler Creek, ca. 2.7 SE of Sandstone Rapids, Coppermine River, 67°25′38.2″N, 115°36′54.9″W ± 50 m, 128 m, 5 July 2014, Saarela, Sokoloff & Bull 3408 (CAN); confluence of Coppermine River and Melville Creek, just W of Coppermine Mountains, 67°15′52″N, 115°30′55.3″W ± 350 m, 178–190 m, 7 July 2014, Saarela, Sokoloff & Bull 3529 (CAN, UBC); Kugluk (Bloody Falls) Territorial Park, upper ledges of rocky (gabbro) S-facing cliffs above the start of Bloody Falls (W bank of River), just E of Portage Trail, 67°44′21.7″N, 115°22′42.2″W ± 25 m, 46 m, 14 July 2014, Saarela, Sokoloff & Bull 3950 (ALA, CAN).

Pedicularis lanata Willd. ex Cham. & Schltdl., Fig. S36B—Woolly lousewort | Amphi-Beringian–North American (N)

Recorded previously from Kugluktuk (Cody, 1954b; Porsild & Cody, 1980). We made collections at Fockler Creek, Melville Creek and Kugluk (Bloody Falls) Territorial Park. Widespread throughout the Canadian Arctic (Porsild & Cody, 1980; Korol, 1992; Cody, Reading & Line, 2003; Aiken et al., 2007; Saarela et al., 2013a).

Specimens Examined: Canada. Nunavut: Kitikmeot Region: Coppermine [Kugluktuk], 67°49′36″N, 115°5′36″W, 14 June 1951, W. I. Findlay 13 (DAO-179422 01-01000622681); Coppermine [Kugluktuk], 67°49′36″N, 115°5′36″W, 3 July 1951, W. I. Findlay 81 (DAO-179423 01-01000622684); Kugluktuk, rocky slopes of North Hill, 67°49′29.6″N, 115°6′31″W ± 50 m, 50 m, 29 June 2014, Saarela, Sokoloff & Bull 3058 (CAN); NW-facing slope above tributary of Fockler Creek, ca. 2.4 km SSW of Sandstone Rapids, Coppermine River, 67°25′46″N, 115°38′49.4″W ± 50 m, 149 m, 3 July 2014, Saarela, Sokoloff & Bull 3279 (CAN, UBC); confluence of Coppermine River and Melville Creek, just W of Coppermine Mountains, 67°15′52″N, 115°30′55.3″W ± 350 m, 178–190 m, 7 July 2014, Saarela, Sokoloff & Bull 3528 (CAN); Kugluk (Bloody Falls) Territorial Park, flats above boardwalk W of Bloody Falls, 67°44′34.5″N, 115°22′27″W ± 100 m, 135 m, 14 July 2014, Saarela, Sokoloff & Bull 4067 (CAN); Kugluk (Bloody Falls) Territorial Park, wet meadow between Coppermine River and large sand hills on W side of river, 0.5 km W of Bloody Falls, 67°44′44.8″N, 115°22′48.3″W ± 15 m, 33 m, 15 July 2014, Saarela, Sokoloff & Bull 4051 (CAN).

Pedicularis langsdorffii subsp. arctica (R. Br.) Pennell ex Hultén, Fig. S37—Arctic lousewort | Asian (NE)–Amphi-Beringian–North American (N)

Recorded previously from Kugluktuk, as Pedicularis arctica R. Br. (Cody, 1954b; Porsild & Cody, 1980), a taxon now recognised as a subspecies of Pedicularis langsdorffii Fisch. ex Steven (Elven et al., 2011). We made collections at Fockler Creek, Kugluk (Bloody Falls) Territorial Park and along the Coppermine River between Muskox Rapids and Escape Rapids. Widespread throughout the central and western Canadian Arctic Archipelago, and recorded from Arctic sites on mainland Nunavut and Northwest Territories (Porsild & Cody, 1980; Cody & Reading, 2005; Aiken et al., 2007; Saarela et al., 2013a). Elven et al. (2011) recognised one other subspecies and one variety in the Panarctic flora, but only subsp. arctica occurs in the Canadian Arctic.

Specimens Examined: Canada. Nunavut: Kitikmeot Region: Coppermine [Kugluktuk], 67°49′36″N, 115°5′36″W, 3 July 1951, W. I. Findlay 78 (DAO-179263 01-01000620014); Coppermine [Kugluktuk], Cemetery Island, 67°50′N, 115°7′W, 14 July 1951, W. I. Findlay 123 (DAO-179262 01-01000616607); sedge meadow at S end of small lake, on flats NW of Fockler Creek, ca. 1.9 km SSE of Sandstone Rapids, Coppermine River, 67°26′1.8″N, 115°37′30.5″W ± 20 m, 170 m, 2 July 2014, Saarela, Sokoloff & Bull 3235 (CAN, UBC); sedge meadow adjacent to small lake on flats N of Fockler Creek, ca. 1.5 km SSE of Sandstone Rapids, Coppermine River, 67°26′8.8″N, 115°37′35.9″W ± 20 m, 168 m, 2 July 2014, Saarela, Sokoloff & Bull 3230b (CAN); S-facing slopes on W side of Coppermine River, about halfway between Escape Rapids and Muskox Rapids, 67°31′18.2″N, 115°36′20.1″W ± 150 m, 115 m, 8 July 2014, Saarela, Sokoloff & Bull 3618 (ALA, CAN); Kugluk (Bloody Falls) Territorial Park, W side of Coppermine River, along ATV trail below slope of sand hill just below picnic bench/lookout area, 67°44′41.5″N, 115°22′14.9″W ± 15 m, 15 m, 17 July 2014, Saarela, Sokoloff & Bull 4152 (CAN); Kugluk (Bloody Falls) Territorial Park, flats above boardwalk W of Bloody Falls, 67°44′34.5″N, 115°22′27″W ± 100 m, 135 m, 18 July 2014, Saarela, Sokoloff & Bull 4154 (CAN).

Pedicularis lapponica L.—Lapland lousewort | Circumpolar-alpine

Recorded previously from Bloody Falls and Kugluktuk (Cody, 1954b; Porsild & Cody, 1980). We made collections at Fockler Creek, Kugluk (Bloody Falls) Territorial Park and Kugluktuk. Elsewhere in the Canadian Arctic recorded from Southampton Island, southern Baffin Island, mainland Nunavut and Northwest Territories, and northern Quebec and Labrador (Porsild & Cody, 1980; Korol, 1992; Cody & Reading, 2005; Aiken et al., 2007). Not recorded from beyond the mainland in the western Canadian Arctic.

Specimens Examined: Canada. Nunavut: Kitikmeot Region: Bloody Falls, 67°44′N, 115°23′W, 18 July 1951, W. I. Findlay 134 (DAO-179454 01-01000619948); Coppermine [Kugluktuk], 67°49′36″N, 115°5′36″W, 13 July 1951, W. I. Findlay 116 (DAO-179452 01-01000619949, QFA0274332); Coppermine [Kugluktuk], 67°49′36″N, 115°5′36″W, 3 July 1951, W. I. Findlay 79 (DAO-179453 01-01000619947); Coppermine [Kugluktuk], 67°49′36″N, 115°5′36″W, 20 June 1951, W. I. Findlay 20 (DAO-179451 01-01000619950); Kugluktuk, upper tundra slope overlooking Coppermine River [67°49′36″N, 115°5′36″W], 16 July 2000, L. K. Benjamin s.n. (ACAD-ECS015869); Kugluktuk, rocky slopes of North Hill, 67°49′29.6″N, 115°6′31″W ± 50 m, 50 m, 29 June 2014, Saarela, Sokoloff & Bull 3056 (CAN, UBC); flats on W side of Fockler Creek, above spruce forest in creek valley, ca. 2.2 km S of Sandstone Rapids, Coppermine River, 67°25′49″N, 115°37′55″W ± 50 m, 152 m, 1 July 2014, Saarela, Sokoloff & Bull 3135 (CAN); Kugluk (Bloody Falls) Territorial Park, day-use area above Bloody Falls (at outhouse and fire pit), 67°44′36.8″N, 115°22′11.1″W ± 25 m, 28 m, 12 July 2014, Saarela, Sokoloff & Bull 3837 (ALA, CAN).

Papaveraceae [1/1]

Papaver hultenii Knaben, Fig. S38—Hulten’s poppy | Beringian American (or amphi-Beringian)

Recorded previously from Kugluktuk, as Papaver radicatum Rottb. (Cody, 1954b; Porsild & Cody, 1980). We made collections at Fockler Creek, south of Escape Rapids, Coppermine Mountains, Kugluk (Bloody Falls) Territorial Park and Kugluktuk. The taxonomy of Arctic poppies is notoriously difficult (Solstad, 2009; Elven et al., 2011). Porsild & Cody (1980) recognised two species in the study area, Papaver hultenii and Papaver radicatum s.l. Kiger & Murray (1997) treated Papaver hultenii as a synonym of Papaver lapponicum (Tolm.) Nordh. Solstad (2009) recognised Papaver hultenii as the major Beringian component of North American Arctic poppies previously recognised as Papaver lapponicum or Papaver radicatum, based primarily on amplified fragment length polymorphism data that did not support a close relationships between Papaver hultenii and Papaver lapponicum. According to their circumscription, Papaver hultenii is the dominant poppy in the western Canadian Arctic, distributed at least as far east as Bathurst Inlet and Cambridge Bay, Nunavut (specimens at CAN det. Solstad & Elven, 2009). Papaver hultenii was described from plants grown from seed collected in the study area at “outlet of Copper Mine River” by M. Hammer in 1948 (holotype at O) (Knaben, 1959). The locality was erroneously recorded in the protologue as being in Alaska. The type specimen (not seen) is reportedly a luxurious specimen that does not resemble wild plants (Solstad, 2009), probably because it is was cultivated rather than wild-collected. Despite apparent molecular differences, distinguishing characters between Papaver hultenii and Papaver lapponicum are unclear. The characters in the key in Solstad (2009) leading to Papaver lapponicum s.l. (blades not divided to mid axis; stigmatic papillae short, little conspicuous) and Papaver hultenii (blades divided to or nearly to mid axis; stigmatic papillae usually long, conspicuous) are difficult to apply. They appear to be highly variable within and among each taxon, based on examination of CAN specimens identified as each taxon by H. Solstad and R. Elven, and do not seem to reliably distinguish them (J. M. Saarela, 2016, personal observation). Two previous collections from the study area were determined as Papaver hultenii by H. Solstad and R. Elven (det. 2009), and we tentatively include all of our poppy collections here. Definitive identification awaits a full taxonomic treatment with keys and morphological descriptions for all North American Arctic poppies.

Specimens Examined: Canada. Nunavut: Kitikmeot Region: Coppermine [Kugluktuk], vicinity of post, near houses [67°49′36″N, 115°5′36″W ± 1.5 km], 26 July 1949, A. E. Porsild 17175 (CAN-127777); Coppermine River, Fort Hearne–Bloody Falls [67.7761972°N, 115.2037222°W ± 7.5 km], 1931, A. M. Berry 7 (CAN-59606); Coppermine River, Fort Hearne–Bloody Falls [67.7761972°N, 115.2037222°W ± 7.5 km], 1931, A. M. Berry 6 (CAN-59607); Coppermine [Kugluktuk], 67°49′36″N, 115°5′36″W, W. I. Findlay 117 (DAO-184610, det. R. Elven & H. Solstad); Coppermine [Kugluktuk], 67°49′36″N, 115°5′36″W, 4 July 1951, W. I. Findlay 88 (DAO-184611, det. R. Elven & H. Solstad); old riverbed of Fockler Creek, E side of Coppermine River, ca. 2.3 km SSE of Sandstone Rapids, Coppermine River, 67°25′45.7″N, 115°37′21.8″W ± 25 m, 166 m, 6 July 2014, Saarela, Sokoloff & Bull 3451 (ALTA, CAN, MO, MT, O); S of Fockler Creek, above small tributary of Fockler Creek, ca. 2.3 km S of Sandstone Rapids, Coppermine River, 67°25′46.3″N, 115°38′2.5″W ± 100 m, 156 m, 6 July 2014, Saarela, Sokoloff & Bull 3454 (CAN, UBC); S-facing slopes on W side of Coppermine River, about halfway between Escape Rapids and Muskox Rapids, 67°31′18.2″N, 115°36′20.1″W ± 150 m, 115 m, 8 July 2014, Saarela, Sokoloff & Bull 3630 (CAN); flats atop and upper slopes of Coppermine Mountains, N/W side of Coppermine River, 67°14′49.9″N, 115°38′43.7″W ± 200 m, 467 m, 9 July 2014, Saarela, Sokoloff & Bull 3777 (CAN); Kugluk (Bloody Falls) Territorial Park, N-facing slopes of large mountain just S of start of Bloody Falls, W side of Coppermine River, 67°44′7.7″N, 115°23′30.4″W ± 15 m, 90 m, 14 July 2014, Saarela, Sokoloff & Bull 4005 (CAN); grassy vacant lot in Kugluktuk, 67°49′30.5″N, 115°5′29.3″W ± 15 m, 21 m, 24 July 2014, Saarela, Sokoloff & Bull 4345 (ALA, CAN).

Parnassiaceae [1/2]

Parnassia kotzebuei Cham. ex Spreng., Fig. S39—Kotzebue’s grass-of-Parnassus | Amphi-Beringian–North American (N)

Recorded previously from Bloody Falls and Kugluktuk (Cody, 1954b; Porsild & Cody, 1980). We made collections at Fockler Creek, Kugluk (Bloody Falls) Territorial Park, Escape Rapids and Richardson Bay. Widespread across the mainland Canadian Arctic, and in the Arctic Archipelago known from Banks and Victoria islands, and southern Baffin Island (Porsild & Cody, 1980; Korol, 1992; Cody, Reading & Line, 2003; Aiken et al., 2007; Saarela et al., 2013a; Blondeau, 2015c).

Specimens Examined: Canada. Nunavut: Kitikmeot Region: Bloody Falls, 67°44′N, 115°23′W, 18 July 1951, W. I. Findlay 142 (DAO-76873 01-01000619906); Coppermine [Kugluktuk], 67°49′36″N, 115°5′36″W, 10 July 1951, W. I. Findlay 107a (DAO-76872 01-01000619903); Coppermine [Kugluktuk], 67°49′36″N, 115°5′36″W, 24 July 1951, W. I. Findlay 172 (ALTA-VP-10154, DAO-76874 01-01000619905, UBC-V40781); Coppermine [Kugluktuk], Cemetery Island, 67°50′N, 115°7′W, 14 July 1951, W. I. Findlay 120 (DAO-76871 01-01000619914); Coppermine [Kugluktuk] [67°49′36″N, 115°5′36″W ± 1.5 km], 10 July 1958, R. D. Wood s.n. (CAN-265526); Coronation Gulf Region, Coppermine River, east bank, 10 July 1955, R. E. Miller 90 (CAN-241984); Coppermine [Kugluktuk], 67°49′36″N, 115°5′36″W, 2 August 1995, T. Dolman 93 (LEA); S of Fockler Creek, along small tributary that runs into Fockler Creek, ca. 2.3 km S of Sandstone Rapids, Coppermine River, 67°25′44.9″N, 115°38′25.9″W ± 100 m, 152 m, 3 July 2014, Saarela, Sokoloff & Bull 3260 (CAN); S-facing slopes above Coppermine River and below spruce forest, ca. 7.8 km NNE of Sandstone Rapids, 67°31′16.2″N, 115°36′52.1″W ± 200 m, 110 m, 8 July 2014, Saarela, Sokoloff & Bull 3650 (CAN); Richardson Bay, confluence of Richardson and Rae rivers at Coronation Gulf, ca. 20 km WNW of Kugluktuk, 67°54′11.2″N, 115°32′27.4″W ± 200 m, 0 m, 8 July 2014, Saarela, Sokoloff & Bull 3680 (CAN, UBC); SE-facing slopes above Escape Rapids, W side of Coppermine River, 67°36′58.7″N, 115°29′18.3″W ± 99 m, 50 m, 8 July 2014, Saarela, Sokoloff & Bull 3733 (CAN); Kugluk (Bloody Falls) Territorial Park, rocky valley immediately SW of Bloody Falls, along rough marked section of Portage Trail, 67°44′34″N, 115°22′16″W ± 50 m, 20 m, 13 July 2014, Saarela, Sokoloff & Bull 3869 (CAN); Kugluk (Bloody Falls) Territorial Park, rocky beach above Bloody Falls, W bank of Coppermine River, 67°44′18″N, 115°22′57.3″W ± 250 m, 34 m, 14 July 2014, Saarela, Sokoloff & Bull 3970 (CAN).

Parnassia palustris subsp. neogaea Hultén, Fig. 67—Grass-of-Parnassus | Amphi-Beringian–North American

Figure 67 Parnassia palustris subsp. neogaea.

(A) Habit, Saarela et al. 4115. (B) Inflorescence, Saarela et al. 4115. Photographs by R. D. Bull.

Recorded previously from the mouth of the Rae River (Porsild & Cody, 1980). We made collections in Kugluk (Bloody Falls) Territorial Park and Kugluktuk. This is primarily a boreal taxon known from other Arctic sites on mainland Nunavut and Northwest Territories, and northern Quebec and Labrador, but not recorded from the Canadian Arctic Archipelago (Porsild & Cody, 1980; Korol, 1992; Cody, Reading & Line, 2003; Aiken et al., 2007; Saarela et al., 2013a; Blondeau, 2015c). It is at the northern edge of its known range in the study area. Taxonomy follows Elven et al. (2011), who recognise two subspecies; the other one, subsp. palustris, is not recorded for Canada. Porsild & Cody (1980) treated it as Parnassia palustris var. neogaea Fernald.

Specimens Examined: Canada. Nunavut: Kitikmeot Region: Rae River mouth [67.919444°N, 115.525°W], 2 August 1955, R. E. Miller 307 (CAN-241985); Kugluk (Bloody Falls) Territorial Park, rocky valley immediately SW of Bloody Falls, along rough marked section of Portage Trail, 67°44′34″N, 115°22′16″W ± 50 m, 20 m, 13 July 2014, Saarela, Sokoloff & Bull 3862 (ALA, CAN); Kugluk (Bloody Falls) Territorial Park, rocky beach above Bloody Falls, W bank of Coppermine River, 67°44′18″N, 115°22′57.3″W ± 250 m, 34 m, 14 July 2014, Saarela, Sokoloff & Bull 3968 (CAN); Kugluk (Bloody Falls) Territorial Park, sandy NE-facing slope above small creek in deep gully, about 0.5 km W of Bloody Falls, 67°44′36.6″N, 115°22′59.3″W ± 41 m, 41 m, 15 July 2014, Saarela, Sokoloff & Bull 4017 (CAN); Kugluk (Bloody Falls) Territorial Park, rocky sandy beach just below Bloody Falls, W side of Coppermine River, vicinity of confluence with small creek, beach seasonally flooded, 67°44′54.5″N, 115°22′17.2″W ± 75 m, 9 m, 17 July 2014, Saarela, Sokoloff & Bull 4115 (ALTA, CAN); W of Kugluktuk on tundra flats above Coppermine River, S of 1 Coronation Drive and N of community power plant, 67°49′28.97″N, 115°5′0.2″W ± 100 m, 8 m, 22 July 2014, Saarela, Sokoloff & Bull 4246 (CAN); grassy sand flats on extensive sandy floodplain of Coppermine River, below steep cliff above river and S of Kugluktuk, 67°48′54.3″N, 115°6′9.1″W ± 20 m, 5 m, 26 July 2014, Saarela, Sokoloff & Bull 4414 (CAN, MO, O).

Plantaginaceae [2/3]

Hippuris lanceolata Retz.—Lance-leaved mare’s-tail | Circumpolar | Noteworthy Record

Newly recorded for the study area. We made a single collection on an unnamed island at the mouth of the Coppermine River. The plants were growing in 1–1.5 ft. of water in a small pond. Hippuris taxonomy follows Elven et al. (2011) and Elven, Murray & Solstad (2012). Given previous misunderstanding of species limits in this genus, earlier range maps of the taxa in the Arctic are unreliable (Elven et al., 2011) with the exception of those in Aiken et al. (2007) based upon taxon records from Coats, Baffin, Banks, Ellesmere, Southampton and Victoria islands, as well as one mainland site along Hudson Bay south of Wager Bay. The species is also recorded on the mainland from Tuktut Nogait National Park and vicinity (Saarela et al., 2013a).

Specimens Examined: Canada. Nunavut: Kitikmeot Region: unnamed island just E (ca. 3.3 km) of Kugluktuk at mouth of Coppermine River, 67°49′29.2″N, 115°1′3.2″W ± 50 m, 1 m, 8 July 2014, Saarela, Sokoloff & Bull 3721 (CAN).

Hippuris vulgaris L.—Common mare’s-tail | Circumboreal | Noteworthy Record

Newly recorded for the study area. We made collections at a site between Escape Rapids and Muskox Rapids, Kugluk (Bloody Falls) Territorial Park and Kugluktuk. See additional comments re: Hippuris taxonomy under Hippuris lanceolata. Elsewhere in the Canadian Arctic, Aiken et al. (2007) recorded this aquatic species on Baffin, Devon, Ellesmere, Southampton and Victoria islands, and across the mainland.

Specimens Examined: Canada. Nunavut: Kitikmeot Region: S-facing slopes on W side of Coppermine River, about halfway between Escape Rapids and Muskox Rapids, 67°31′18.2″N, 115°36′20.1″W ± 150 m, 115 m, 8 July 2014, Saarela, Sokoloff & Bull 3616 (ALA, ALTA, CAN, MO, O, UBC); Kugluk (Bloody Falls) Territorial Park, rocky valley immediately SW of Bloody Falls, along rough marked section of Portage Trail, head of small unnamed pond just W of falls, 67°44′42.8″N, 115°22′29.2″W ± 10 m, 9 m, 13 July 2014, Saarela, Sokoloff & Bull 3902 (CAN); flats below large overhanging cliffs above Coppermine River, just S of Kugluktuk, 67°48′56.7″N, 115°6′22.6″W ± 10 m, 2 m, 26 July 2014, Saarela, Sokoloff & Bull 4410 (CAN).

Plantago canescens subsp. richardsonii (Decne) Tzvelev—Hairy plantain, grey-pubescent plantain, arctic plantain | American Beringian

Previously reported from Kugluktuk (Cody, 1954b; Bassett, 1967; Porsild & Cody, 1980). We made collections at Tundra Lake, Melville Creek, Richardson Bay and Kugluk (Bloody Falls) Territorial Park. Elven et al. (2011) recognise three subspecies in Plantago canescens Adams, of which only subsp. richardsonii is present in North America. Elsewhere in the Canadian Arctic recorded from Banks and Victoria islands, mainland Northwest Territories, a site in Nunavut west of the study region, and Bathurst Inlet, the taxon’s known eastern limit (Porsild & Cody, 1980; Aiken et al., 2007; Saarela et al., 2013a).

Specimens Examined: Canada. Nunavut: Kitikmeot Region: Coppermine [Kugluktuk], 67°49′36″N, 115°5′36″W, 24 July 1951, W. I. Findlay 178B (ALTA-VP-15952, DAO-179628 01-01000620139); meadow just S of Tundra Lake, ca. 4.2 km SE of Sandstone Rapids, Coppermine River, 67°25′34.8″N, 115°33′27.8″W ± 20 m, 265 m, 5 July 2014, Saarela, Sokoloff & Bull 3433 (CAN, UBC); confluence of Coppermine River and Melville Creek, just W of Coppermine Mountains, 67°15′52″N, 115°30′55.3″W ± 350 m, 178–190 m, 7 July 2014, Saarela, Sokoloff & Bull 3508 (CAN); S-facing sandstone cliffs above Coppermine River, ca. 7.8 km NNE of Sandstone Rapids, 67°31′15.1″N, 115°36′19.1″W ± 50 m, 106 m, 8 July 2014, Saarela, Sokoloff & Bull 3637 (ALA, CAN); Richardson Bay, confluence of Richardson and Rae rivers at Coronation Gulf, ca. 20 km WNW of Kugluktuk, 67°54′11.2″N, 115°32′27.4″W ± 200 m, 0 m, 8 July 2014, Saarela, Sokoloff & Bull 3666 (CAN, MT, O); Kugluk (Bloody Falls) Territorial Park, rocky cliffs and ledges directly above (W side) of Bloody Falls, just S of heavily used day-use/fishing area, 67°44′40.1″N, 115°22′4.9″W ± 20 m, 8 m, 12 July 2014, Saarela, Sokoloff & Bull 3810 (ALTA, CAN).

Plumbaginaceae [1/1]

Armeria maritima subsp. sibirica (Turcz. ex Boiss.) Nyman, Fig. S40—Arctic thrift, sea pink | Circumpolar

Recorded previously from Kugluktuk (Cody, 1954b; Porsild & Cody, 1980). We made collections at Fockler Creek, Big Creek, Richardson Bay, Kugluk (Bloody Falls) Territorial Park and near Heart Lake. Taxonomy follows Lefèbvre & Vekemans (1995, 2005), who recognise four subspecies of Armeria maritima (Mill.) Willd. s.l. in North America, of which two occur in the Arctic. The subspecies sibirica is widely distributed in the Canadian Arctic, while subsp. maritima is restricted to Greenland. Elven et al. (2011) recognised this taxon at species level, as Armeria scabra Pall. ex Roem. & Schult. Porsild & Cody (1980) recognised two subspecies in Armeria maritima, the Amphi-Beringian subsp. arctica (Cham.) Hultén and the Amphi-Atlantic subsp. labradorica (Wallr.) Hultén. Both of these were included in subsp. sibirica by Lefèbvre (1995) and Lefèbvre & Vekemans (2005).

Specimens Examined: Canada. Nunavut: Kitikmeot Region: Coppermine River, Fort Hearne–Bloody Falls [67.7761972°N, 115.2037222°W ± 7.5 km], 1931, A. M. Berry 20 (CAN-91995); Coppermine [Kugluktuk], 67°49′36″N, 115°5′36″W, 3 July 1951, W. I. Findlay 77 (DAO1178656 01-01000619661); Coppermine [Kugluktuk], 67°49′36″N, 115°5′36″W, 4 August 1951, W. I. Findlay 249 (ACAD-30916, ALTA-VP-14343, DAO-178655 01-01000619660); Coppermine [Kugluktuk], Cemetery Island [67.834275°N, 115.0671833°W ± 0.8 km], 9 July 1955, R. E. Miller 64 (CAN-242003); Coppermine [Kugluktuk], vic. of hamlet and airstrip, 67.78°N, 115.5°W ± 3,615 m, 23 June 1999, C. L. Parker & I. Jonsdottir 9097 (ALA); flats on W side of Fockler Creek, above spruce forest in creek valley, ca. 2.2 km S of Sandstone Rapids, Coppermine River, 67°25′49″N, 115°37′55″W ± 50 m, 152 m, 1 July 2014, Saarela, Sokoloff & Bull 3110 (ALA, CAN); old riverbed of Fockler Creek, ca. 2.3 km SSE of Sandstone Rapids, Coppermine River, 67°25′45.7″N, 115°37′21.8″W ± 25 m, 166 m, 2 July 2014, Saarela, Sokoloff & Bull 3186 (CAN); S of Fockler Creek, along small tributary that runs into Fockler Creek, ca. 2.3 km S of Sandstone Rapids, Coppermine River, 67°25′44.9″N, 115°38′25.9″W ± 100 m, 152 m, 3 July 2014, Saarela, Sokoloff & Bull 3251 (ALTA, CAN); forest and slopes at confluence of Big Creek and Coppermine River, N side of Coppermine River, S side of Coppermine Mountains, 67°14′29.3″N, 116°2′44.5″W ± 250 m, 180–199 m, 7 July 2014, Saarela, Sokoloff & Bull 3559 (CAN, O); Richardson Bay, confluence of Richardson and Rae rivers at Coronation Gulf, ca. 20 km WNW of Kugluktuk, 67°54′11.2″N, 115°32′27.4″W ± 200 m, 0 m, 8 July 2014, Saarela, Sokoloff & Bull 3690 (CAN); unnamed island just E (ca. 3.3 km) of Kugluktuk at mouth of Coppermine River, 67°49′29.2″N, 115°1′3.2″W ± 50 m, 1 m, 8 July 2014, Saarela, Sokoloff & Bull 3709 (CAN, UBC, US); Kugluk (Bloody Falls) Territorial Park, flats on top of mountain on W side of Coppermine River, just S of the start of Bloody Falls Rapids, 67°44′2.8″N, 115°23′39.3″W ± 250 m, 110 m, 14 July 2014, Saarela, Sokoloff & Bull 3988 (CAN, MO); hummocky tundra just SW of sewage retaining pond, N side of road to Heart Lake cemetery, 5.4 km SW of mouth of Coppermine River, 67°48′44.6″N, 115°12′27.2″W ± 3 m, 35 m, 23 July 2014, Saarela, Sokoloff & Bull 4327 (CAN, MT).

Polygonaceae [5/5]

Bistorta vivipara (L.) Delarbre, Fig. 68A—Alpine bistort | Circumboreal–polar

Figure 68 Bistorta vivipara and Koenigia islandica.

Bistorta vivipara: (A) habit, Kugluk (Bloody Falls) Territorial Park, Nunavut, 17 July 2014. Koenigia islandica: (B) habit, Saarela et al. 4315. Photographs by P. C. Sokoloff.

Recorded previously from Kugluktuk, as Polygonum viviparum L. (Cody, 1954b; Porsild & Cody, 1980). We made collections at Fockler Creek, Kendall River, Kugluk (Bloody Falls) Territorial Park and Kugluktuk. Widespread throughout the Canadian Arctic (Porsild & Cody, 1980; Cody, Scotter & Zoltai, 1989; Korol, 1992; Cody & Reading, 2005; Aiken et al., 2007; Saarela et al., 2013a; Blondeau, 2015d).

Specimens Examined: Canada. Nunavut: Kitikmeot Region: Coppermine [Kugluktuk], 67°49′36″N, 115°5′36″W, 24 July 1951, W. I. Findlay 171 (DAO-189356 01-01000677353); S of Fockler Creek, along small tributary that runs into Fockler Creek, ca. 2.3 km S of Sandstone Rapids, Coppermine River, 67°25′44.9″N, 115°38′25.9″W ± 100 m, 152 m, 3 July 2014, Saarela, Sokoloff & Bull 3269 (CAN); slopes on E side of Coppermine River, N of its confluence with Fockler Creek, ca. 0.8 km SW of Sandstone Rapids, 67°26′36.9″N, 115°38′50.1″W ± 50 m, 128 m, 4 July 2014, Saarela, Sokoloff & Bull 3379 (CAN, UBC); confluence of Coppermine and Kendall rivers (NW side of Coppermine River, S side of Kendall River), 67°6′51.1″N, 116°8′18.3″W ± 150 m, 220 m, 7 July 2014, Saarela, Sokoloff & Bull 3579 (ALA, CAN); Kugluk (Bloody Falls) Territorial Park, rocky cliffs and ledges directly above (W side) of Bloody Falls, just S of heavily used day-use/fishing area, 67°44′40.1″N, 115°22′4.9″W ± 20 m, 8 m, 13 July 2014, Saarela, Sokoloff & Bull 3926 (CAN); Kugluk (Bloody Falls) Territorial Park, W side of Coppermine River, between Sandy Hills and Bloody Falls, 67°45′17.6″N, 115°22′14.2″W ± 20 m, 76 m, 17 July 2014, Saarela, Sokoloff & Bull 4141 (CAN); Kugluktuk, roadside and flats between buildings, 67°49′27.4″N, 115°5′26.2″W ± 25 m, 29 m, 26 July 2014, Saarela, Sokoloff & Bull 4395 (ALTA, CAN).

Koenigia islandica L., Fig. 68B—Iceland purslane | Circumpolar-alpine

Previously recorded from Kugluktuk (Porsild & Cody, 1980; Hedberg, 1997). We made collections along the northern shore of Heart Lake, where it was locally common. This diminutive annual taxon has a scattered distribution across mainland Nunavut (Porsild & Cody, 1980; Cody, Scotter & Zoltai, 1989; Cody, Reading & Line, 2003), but is likely more common than existing collections suggest. Elsewhere in the Canadian Arctic recorded from Baffin, Devon and Southampton islands, and northern Quebec and Labrador (Aiken et al., 2007; Blondeau, 2015d).

Specimens Examined: Canada. Nunavut: Kitikmeot Region: Coppermine [Kugluktuk] [67°49′36″N, 115°5′36″W ± 1.5 km], 10 July 1955, R. E. Miller 80 (CAN-241968); N side of Heart Lake, below rocky cliff, SW of Kugluktuk, 5.64 km SW of mouth of Coppermine River, 67°48′33.4″N, 115°12′38.8″W ± 25 m, 31 m, 23 July 2014, Saarela, Sokoloff & Bull 4311 (CAN); N side of Heart Lake, below rocky cliff, SW of Kugluktuk, 5.64 km SW of mouth of Coppermine River, 67°48′33.8″N, 115°12′52.9″W ± 15 m, 39 m, 23 July 2014, Saarela, Sokoloff & Bull 4315 (CAN, UBC).

Oxyria digyna (L.) Hill—Mountain sorrel, alpine sorrel | Circumpolar–alpine

Recorded previously from Kugluktuk (Cody, 1954b; Porsild & Cody, 1980). We made collections at Fockler Creek, Kugluk (Bloody Falls) Territorial Park and Kugluktuk. Widely distributed throughout the Canadian Arctic (Porsild & Cody, 1980; Cody, Scotter & Zoltai, 1989; Korol, 1992; Aiken et al., 2007; Saarela et al., 2013a; Blondeau, 2015d).

Specimens Examined: Canada. Nunavut: Kitikmeot Region: Coppermine [Kugluktuk], 67°49′36″N, 115°5′36″W, 2 July 1951, W. I. Findlay 67 (DAO-186198 01-01000677369); Kugluktuk, 67.798017°N, 115.23075°W, 22 June 2006, J. Davis 605 (CAN-597645); Kugluktuk, disturbed rocky/gravelly ground around house, top of NW-facing gravel slope, just SE of Igalik Building, 67°49′29.6″N, 115°6′31″W ± 10 m, 13 m, 30 June 2014, Saarela, Sokoloff & Bull 3100 (ALA, ALTA, CAN); spruce forest along Fockler Creek, ca. 2.3 km SSE of Sandstone Rapids, Coppermine River, 67°25′45.7″N, 115°37′21.8″W ± 25 m, 166 m, 2 July 2014, Saarela, Sokoloff & Bull 3189 (CAN); Kugluk (Bloody Falls) Territorial Park, rocky valley immediately SW of Bloody Falls, along rough marked section of Portage Trail, 67°44′34″N, 115°22′16″W ± 50 m, 20 m, 13 July 2014, Saarela, Sokoloff & Bull 3888 (CAN); Kugluk (Bloody Falls) Territorial Park, rocky valley immediately SW of Bloody Falls, along rough marked section of Portage Trail, 67°44′34″N, 115°22′16″W ± 50 m, 20 m, 18 July 2014, Saarela, Sokoloff & Bull 4162 (CAN); N-facing slope on high terrace above Bloody Falls rapids, SE side of Coppermine River, 67°44′16.2″N, 115°22′1.8″W ± 25 m, 91 m, 19 July 2014, Saarela, Sokoloff & Bull 4227 (CAN, MO, MT); SE edge of Kugluktuk, rocky cliffs overlooking Coppermine River, 67°49′9.2″N, 115°5′40.4″W ± 50 m, 28 m, 24 July 2014, Saarela, Sokoloff & Bull 4353 (CAN, UBC).

Polygonum aviculare L. s.l.—Prostrate knotweed | Noteworthy Record

First record for the study area and Nunavut. The species was found growing in patches on disturbed ground along a roadside in Kugluktuk. The introduced Polygonum aviculare is difficult to distinguish from the native Polygonum humifusum subsp. caurianum (B. L. Rob.) Costea & Tardif (recognised as Polygonum caurianum B. L. Rob. in Porsild & Cody (1980)), as the ranges of most characteristics used to distinguish the species overlap (Costea & Tardif, 2003). The main character used to distinguish them in the key in Costea, Tardif & Hinds (2003) deals with the morphology of the achene (striate-tubercled, uniformly tubercled, or obscurely tubercled vs. smooth or roughened; these character states are illustrated in Wolf & McNeill (1986)), which must be observed at very high magnification. The character states can be difficult to discern (J. M. Saarela, 2016, personal observation). We tentatively assign our collection to Polygonum aviculare, on the basis of its minutely papillate achenes (visible at 100×) [vs. smooth or roughened], green blades [vs. usually reddish or purple tinged], perianth 2.7–3.2 (–3.4) mm (vs. 1.5–2.3 (–3) mm), achenes ≥2.3 mm long (vs. 1.4–1.6 (–2.2) mm), leaves alternate at proximal nodes (vs. often opposite at proximal nodes), leaves ocrea usually ≥4 mm (vs. 2–3 (–4) mm) and pedicels ocrea ca. 4 mm (vs. 0.5–1.5 mm).

Six subspecies are recognised in Polygonum aviculare (Costea, Tardif & Hinds, 2003). None are reported for Nunavut in Costea, Tardif & Hinds (2003) and Costea & Tardif (2005), and subsp. aviculare, subsp. buxiforme (Small) Costea & Tardif and subsp. depressum (Meisn.) Arcang. are recorded for Northwest Territories (though the latter not recorded for Northwest Territories in Costea & Tardif (2005)). There is a single record of subsp. depressum from Arctic Quebec (Blondeau, 2015d). Our plants fit subsp. buxiforme in most characteristics, but do not have outer tepals pouched at the base, a characteristic of this taxon. Subspecies buxiforme is thought to be native to North American origin (Costea & Tardif, 2003) and is the northernmost infraspecific taxon in Canada, known from as far north as the Mackenzie Delta. The nearest known location to our Kugluktuk collection is from western Great Bear Lake (Costea & Tardif, 2005). Polygonum aviculare was treated, but not mapped or illustrated, in Porsild & Cody (1980). If confirmed, this would be the first report of Polygonum aviculare subsp. buxiforme for Nunavut.

Specimens Examined: Canada. Nunavut: Kitikmeot Region: dry roadside in Kugluktuk and in wet ditch, 67°49′29.3″N, 115°5′25.3″W ± 3 m, 29 m, 26 July 2014, Saarela, Sokoloff & Bull 4391 (ALA, ALTA, CAN, UBC).

Rumex arcticus Trautv., Fig. 69—Arctic dock | European (NE)–Asian (N)–amphi-Beringian

Figure 69 Rumex arcticus.

(A) Habit, Saarela et al. 3925. (B) Inflorescence, Saarela et al. 3925. Photographs R. D. Bull.

Recorded previously from Bloody Falls and Kugluktuk (Cody, 1954b; Porsild & Cody, 1980). We made collections at Fockler Creek, Tundra Lake, Kugluk (Bloody Falls) Territorial Park and Heart Lake. In Nunavut this species reaches its known eastern limit at Bathurst Inlet (Porsild & Cody, 1980) and along the Hood River (Gould & Walker, 1997). A more easterly collection from Churchill requires confirmation (Porsild & Cody, 1980; Mosyakin, 2005). Elsewhere in the Canadian Arctic recorded from Banks Island, Tuktut Nogait National Park and vicinity, and the northwestern mainland (Porsild & Cody, 1980; Aiken et al., 2007; Saarela et al., 2013a).

Specimens Examined: Canada. Nunavut: Kitikmeot Region: Coppermine River, Fort Hearne–Bloody Falls [67.7761972°N, 115.2037222°W ± 7.5 km], 1931, A. M. Berry 3 (CAN-43376); Coppermine [Kugluktuk] [67°49′36″N, 115°5′36″W ± 1.5 km], 12 July 1958, R. D. Wood s.n. (CAN-265577); Bloody Falls, 67°44′N, 115°23′W, 18 July 1951, W. I. Findlay 146 (DAO-186080 01-01000677365); W shore of Tundra Lake, ca. 4.3 km SE of Sandstone Rapids, Coppermine River, 67°25′39.2″N, 115°33′11.5″W ± 5 m, 252 m, 5 July 2014, Saarela, Sokoloff & Bull 3440 (CAN, UBC); S of Fockler Creek, above small tributary of Fockler Creek, ca. 2.3 km S of Sandstone Rapids, Coppermine River, 67°25′46.3″N, 115°38′2.5″W ± 100 m, 156 m, 6 July 2014, Saarela, Sokoloff & Bull 3452 (ALA, ALTA, CAN); S-facing slopes on W side of Coppermine River, about halfway between Escape Rapids and Muskox Rapids, 67°31′18.2″N, 115°36′20.1″W ± 150 m, 115 m, 8 July 2014, Saarela, Sokoloff & Bull 3622 (CAN); Kugluk (Bloody Falls) Territorial Park, flats above boardwalk W of Bloody Falls, 67°44′34.5″N, 115°22′27″W ± 100 m, 135 m, 13 July 2014, Saarela, Sokoloff & Bull 3925 (CAN, MO, O); N side of Heart Lake, below rocky cliff, SW of Kugluktuk, 5.64 km SW of mouth of Coppermine River, 67°48′33.4″N, 115°12′38.8″W ± 25 m, 31 m, 23 July 2014, Saarela, Sokoloff & Bull 4312 (CAN).

Primulaceae [2/4]

Androsace chamaejasme subsp. andersonii (Hultén) Hultén, Fig. S41—Rock jasmine | Asian (N/C)–amphi-Beringian–North American (NW)

Recorded previously from Kugluktuk (Cody, 1954b; Porsild & Cody, 1980), its known eastern limit on mainland Nunavut. We collected the taxon along Richardson Bay. Elsewhere in the Canadian Arctic recorded from Banks and Victoria islands, and mainland sites west of the study area (Porsild & Cody, 1980; Aiken et al., 2007; Saarela et al., 2013a). Taxonomy follows Elven et al. (2011). Most authors have recognised the taxon as subsp. lehmanniana (Spreng.) Hultén (Hultén, 1968; Kelso, 2009). However, application of the name lehmanniana is problematic because the type specimen, from the Caucasus and originally housed at B, was destroyed. Elven et al. (2011) argued that the information in the protologue is insufficient to be certain the Caucasus plants are the same race as the Arctic ones, and thus applied the name andersonii (type from Alaska). However, they noted the name lehmanniana may ultimately prove to be correct at subspecific rank for this taxon. The other northern subspecies, subsp. capitata (Willd. ex Roem. & Schult.) Korobkov, is restricted to the Russian Far East (Elven et al., 2011).

Specimens Examined: Canada. Nunavut: Kitikmeot Region: Coppermine [Kugluktuk] [67°49′36″N, 115°5′36″W ± 1.5 km], 29 June 1958, R. D. Wood s.n. (CAN-265489); Richardson Bay, confluence of Richardson and Rae rivers at Coronation Gulf, ca. 20 km WNW of Kugluktuk, 67°54′11.2″N, 115°32′27.4″W ± 200 m, 0 m, 8 July 2014, Saarela, Sokoloff & Bull 3679 (CAN, UBC).

Androsace septentrionalis L.—Northern fairy-candelabra | Circumboreal-polar

Recorded previously from Kugluktuk (Cody, 1954b; Porsild & Cody, 1980). We made collections at Kugluktuk, Fockler Creek, near Heart Lake cemetery and on an island at the mouth of the Coppermine River. Elsewhere in the Canadian Arctic recorded from Axel Heiberg (northern) Baffin, Banks, Ellesmere, King William, Melville, Southampton and Victoria islands, and some mainland sites to western Hudson Bay (Porsild & Cody, 1980; Korol, 1992; Aiken et al., 2007; Saarela et al., 2013a). Numerous infraspecific taxa have been recognised in this widespread species, but none are recognised in recent treatments (Kelso, 2009; Elven et al., 2011).

Specimens Examined: Canada. Nunavut: Kitikmeot Region: Coppermine [Kugluktuk], vicinity of post [67°49′36″N, 115°5′36″W ± 1.5 km], 26 July 1949, A. E. Porsild 17183 (CAN-128061); above Coppermine River, 3 July 1958, R. D. Wood s.n. (CAN-265487); Kugluktuk, disturbed rocky/gravelly ground around house, top of NW-facing gravel slope, just SE of Igalik Building, 67°49′29.6″N, 115°6′31″W ± 10 m, 13 m, 30 June 2014, Saarela, Sokoloff & Bull 3095 (CAN, UBC); S of Fockler Creek, along small tributary that runs into Fockler Creek, ca. 2.3 km S of Sandstone Rapids, Coppermine River, 67°25′44.9″N, 115°38′25.9″W ± 100 m, 152 m, 3 July 2014, Saarela, Sokoloff & Bull 3274 (CAN); E side of Fockler Creek, in valley just above creek’s confluence with the Coppermine River, ca. 1.4 km SSW of Sandstone Rapids, 67°26′21.4″N, 115°38′54″W ± 5 m, 140 m, 4 July 2014, Saarela, Sokoloff & Bull 3358 (CAN); unnamed island just E (ca. 3.3 km) of Kugluktuk at mouth of Coppermine River, 67°49′29.2″N, 115°1′3.2″W ± 50 m, 1 m, 8 July 2014, Saarela, Sokoloff & Bull 3711 (ALA, CAN); gravel roadside SW of Kugluktuk, S side of road to Heart Lake cemetery, just beyond sewage retention pond, 5.59 km SW of mouth of Coppermine River, 67°48′39″N, 115°12′38.7″W ± 25 m, 46 m, 23 July 2014, Saarela, Sokoloff & Bull 4320 (CAN).

Primula egaliksensis Wormsk., Fig. S42—Greenland primrose | Amphi-Beringian–North American (N)–amphi-Atlantic

Recorded previously from Kugluktuk (Cody, 1954b; Porsild & Cody, 1980). Our collections, from Fockler Creek and Kugluk (Bloody Falls) Territorial Park, fill in a short distribution gap along the Coppermine River between Kugluktuk and Great Bear Lake, where collections have been made previously (Porsild & Cody, 1980). Kelso (1991) reported the habitat for this species as damp silt along stream banks. Four of the five populations we encountered occurred along the banks of the Coppermine River or its tributaries; the fifth was found on the banks of a small pond adjacent to the Coppermine River. Elsewhere in the Canadian Arctic known from mainland Northwest Territories, Nunavut and northern Quebec (Porsild & Cody, 1980; Cody, Scotter & Zoltai, 1984; Korol, 1992; Saarela et al., 2013a). This species may be mistaken for the superficially similar Primula stricta, which also occurs in the western Canadian Arctic (Gillespie et al., 2015).

Specimens Examined: Canada. Nunavut: Kitikmeot Region: Coppermine [Kugluktuk], 67°49′36″N, 115°5′36″W, 13 July 1951, W. I. Findlay 118 (DAO-178772 01-000457230); Coppermine [Kugluktuk], 67°49′36″N, 115°5′36″W, 10 July 1951, W. I. Findlay 110 (DAO-178474 01-000457229); Coppermine [Kugluktuk], along small stream flowing into Coppermine River [67°49′36″N, 115°5′36″W ± 1.5 km], 10 July 1958, R. D. Wood s.n. (CAN-265486); sedge meadow at S end of small lake, on flats NW of Fockler Creek, ca. 1.9 km SSE of Sandstone Rapids, Coppermine River, 67°26′1.8″N, 115°37′30.5″W ± 20 m, 170 m, 2 July 2014, Saarela, Sokoloff & Bull 3234 (CAN); S of Fockler Creek, along small tributary that runs into Fockler Creek, ca. 2.3 km S of Sandstone Rapids, Coppermine River, 67°25′44.9″N, 115°38′25.9″W ± 100 m, 152 m, 3 July 2014, Saarela, Sokoloff & Bull 3254 (CAN); slopes on E side of Coppermine River, N of its confluence with Fockler Creek, ca. 0.8 km SW of Sandstone Rapids, 67°26′36.9″N, 115°38′50.1″W ± 50 m, 128 m, 4 July 2014, Saarela, Sokoloff & Bull 3394 (CAN, UBC); Kugluk (Bloody Falls) Territorial Park, rocky valley immediately SW of Bloody Falls, along rough marked section of Portage Trail, 67°44′34″N, 115°22′16″W ± 50 m, 20 m, 13 July 2014, Saarela, Sokoloff & Bull 3865 (ALA, CAN); wet seepy area above start of Bloody Falls, SE side of Coppermine River, 67°44′17.1″N, 115°22′31″W ± 20 m, 67 m, 19 July 2014, Saarela, Sokoloff & Bull 4225 (ALTA, CAN).

Primula stricta Wormsk.—Coastal primrose, Strict primrose | North American (N)–amphi-Atlantic–European (N)

Recorded previously for the study area (Porsild & Cody, 1980), but we were unable to locate a voucher specimen for confirmation. We made a single collection on an unnamed island at the mouth of the Coppermine River. This species is primarily found in coastal areas (where we encountered it), but sometimes occurs inland along rivers (Kelso, 1991). Elsewhere in the Canadian Arctic known from Banks and Victoria islands, mainland Northwest Territories, a few mainland Nunavut sites and northern Quebec (Porsild & Cody, 1980; Korol, 1992; Aiken et al., 2007; Saarela et al., 2013a).

Specimens Examined: Canada. Nunavut: Kitikmeot Region: unnamed island just E (ca. 3.3 km) of Kugluktuk at mouth of Coppermine River, 67°49′29.2″N, 115°1′3.2″W ± 50 m, 1 m, 8 July 2014, Saarela, Sokoloff & Bull 3716 (CAN, UBC).

Ranunculaceae [5/11]

Anemone parviflora Michx., Figs. S43A and S43B—Northern white anemone, large-flowered anemone | Amphi-Beringian (E)–North American (N)

Recorded previously from Kugluktuk (Cody, 1954b), but not mapped there in Porsild & Cody (1980). We made collections at Fockler Creek and Kugluk (Bloody Falls) Territorial Park. Elsewhere in the Canadian Arctic recorded from Banks and Victoria islands, and some adjacent mainland sites (Porsild & Cody, 1980; Cody & Reading, 2005; Aiken et al., 2007; Saarela et al., 2013a; Blondeau, 2015e).

Specimens Examined: Canada. Nunavut: Kitikmeot Region: Coppermine [Kugluktuk], 67°49′36″N, 115°5′36″W, 7 June 1951, W. I. Findlay 6 (ALTA-VP-8704, DAO-184189 01-01000619894); old riverbed of Fockler Creek, ca. 2.3 km SSE of Sandstone Rapids, Coppermine River, 67°25′45.7″N, 115°37′21.8″W ± 25 m, 166 m, 1 July 2014, Saarela, Sokoloff & Bull 3157 (CAN); Kugluk (Bloody Falls) Territorial Park, rocky valley immediately SW of Bloody Falls, along rough marked section of Portage Trail, 67°44′34″N, 115°22′16″W ± 50 m, 20 m, 13 July 2014, Saarela, Sokoloff & Bull 3890 (ALA, ALTA, CAN, UBC).

Anemone richardsonii Hook., Figs. S43C and S43D—Yellow anemone, Richardson’s anemone | Asian (NE)–amphi-Beringian–North American (N)

Recorded previously from Kugluktuk (Cody, 1954b; Porsild & Cody, 1980). We made collections in Kugluktuk and Kugluk (Bloody Falls) Territorial Park. Elsewhere in the Canadian Arctic recorded from Banks Island and across the mainland (Porsild & Cody, 1980; Cody & Reading, 2005; Aiken et al., 2007; Saarela et al., 2013a; Blondeau, 2015e).

Specimens Examined: Canada. Nunavut: Kitikmeot Region: Coppermine [Kugluktuk], 67°49′36″N, 115°5′36″W, 20 June 1951, W. I. Findlay 21 (DAO-184281 01-01000619895); Kugluktuk, damp tundra slope above Coppermine River [67°49′36″N, 115°5′36″W], 14 June 2000, L. K. Benjamin s.n. (ACAD-ECS015893); Kugluktuk, NW-facing slope across road from coast, 67°49′41.2″N, 115°6′0.7″W ± 5 m, 1 m, 30 June 2014, Saarela, Sokoloff & Bull 3103 (CAN, UBC); old riverbed of Fockler Creek, ca. 2.3 km SSE of Sandstone Rapids, Coppermine River, 67°25′45.7″N, 115°37′21.8″W ± 25 m, 166 m, 1 July 2014, Saarela, Sokoloff & Bull 3155 (CAN); Kugluk (Bloody Falls) Territorial Park, day-use area above Bloody Falls (at outhouse and fire pit), 67°44′36.8″N, 115°22′11.1″W ± 25 m, 28 m, 12 July 2014, Saarela, Sokoloff & Bull 3833 (CAN); SE edge of Kugluktuk, rocky cliffs overlooking Coppermine River, 67°49′9.2″N, 115°5′40.4″W ± 50 m, 28 m, 24 July 2014, Saarela, Sokoloff & Bull 4349 (CAN).

Caltha palustris subsp. radicans (T. F. Forst.) Syme, Fig. 70—Marsh marigold | European (N)–Asian (N)–amphi-Beringian–North American (NW)

Figure 70 Caltha palustris subsp. radicans.

(A) Inflorescence, Saarela et al. 3104. (B) Habit, Saarela et al. 3104. Photographs by P. C. Sokoloff.

Recorded previously from Kugluktuk, as Caltha palustris var. arctica (R. Br.) Huth. (Cody, 1954b; Porsild & Cody, 1980), a heterotypic synonym of subsp. radicans (Elven et al., 2011). Ford (1997) did not recognise infraspecific taxa, while Elven et al. (2011) recognised subsp. radicans as the main Arctic race; we follow the latter treatment. We made two collections and at each site the species was uncommon. Plants grew along the edges of a small pond near the airport terminal in Kugluktuk, and in a marshy area along a creek in Kugluk (Bloody Falls) Territorial Park. Elsewhere in the Canadian Arctic recorded from a few sites on mainland Nunavut, Banks, King William, Melville and Victoria islands, and some mainland Northwest Territories sites (Porsild & Cody, 1980; Cody, 1996a; Gould & Walker, 1997; Cody, Reading & Line, 2003; Cody & Reading, 2005; Aiken et al., 2007; Saarela et al., 2013a).

Specimens Examined: Canada. Nunavut: Kitikmeot Region: Coppermine [Kugluktuk], between R.C. [Roman Catholic] Mission and DOT [67.826667°N, 115.09333°W ± 1.5 km], 1 July 1958, R. D. Wood s.n. (CAN-265556); Coppermine [Kugluktuk], 67°49′36″N, 115°5′36″W, 10 July 1951, W. I. Findlay 112 (ACAD-30932, ALTA-VP-8815, DAO-183976 01-01000616658); Coppermine [Kugluktuk], 67°49′36″N, 115°5′36″W, 28 June 1951, W. I. Findlay 45 (DAO-183977 01-01000616659); Coppermine [Kugluktuk], 67°49′36″N, 115°5′36″W, 6 August 1951, W. I. Findlay 254 (DAO-184499 01-01000616602); Kugluktuk, across Coronation Street from coast, 67°49′42″N, 115°6′6.6″W ± 2 m, 1 m, 30 June 2014, Saarela, Sokoloff & Bull 3104 (ALA, CAN, UBC); Kugluk (Bloody Falls) Territorial Park, rocky valley immediately SW of Bloody Falls, along rough marked section of Portage Trail, head of small unnamed pond just W of falls, 67°44′42.8″N, 115°22′29.2″W ± 10 m, 9 m, 13 July 2014, Saarela, Sokoloff & Bull 3901 (ALTA, CAN, O); flats below large overhanging cliffs above Coppermine River, just S of Kugluktuk, 67°48′56.7″N, 115°6′22.6″W ± 10 m, 2 m, 26 July 2014, Saarela, Sokoloff & Bull 4407 (CAN, US).

Coptidium pallasii (Schltdl.) Tzvelev—Pallas’ buttercup | European (N)–Asian (N)–amphi-Beringian–North American (N)

Recorded previously from Kugluktuk (Cody, 1954b; Porsild & Cody, 1980), where noted to be rare. We did not encounter this conspicuous emergent aquatic species in 2014. Elsewhere on mainland Nunavut known from several sites (Savile & Calder, 1952; Korol, 1992; Cody, 1996a; Cody, Reading & Line, 2003), and elsewhere in the Canadian Arctic known from southern Baffin Island, northern Quebec and Labrador, and some northwestern Northwest Territories sites (Porsild & Cody, 1980; Cody, 1996a; Aiken et al., 2007; Saarela et al., 2013a; Blondeau, 2015e). Previously recognised as Ranunculus pallasii Schltdl.

Specimens Examined: Canada. Nunavut: Kitikmeot Region: Coppermine [Kugluktuk], 67°49′36″N, 115°5′36″W, 6 August 1951, W. I. Findlay 254 (DAO-184499 01-01000616602).

Halerpestes cymbalaria (Pursh) Greene, Fig. 71—Northern seaside buttercup | Asian (N/C) & North American

Figure 71 Halerpestes cymbalaria.

(A) Inflorescence and developing fruits, Saarela et al. 4325. (B) Habit, Saarela et al. 3669. (C) Habitat, Saarela et al. 3669. Photographs by R. D. Bull (A) and J. M. Saarela (B, C).

Recorded previously for the study area (Cody, 1954b; Porsild & Cody, 1980), but we were unable to locate a voucher specimen for confirmation. We made collections at Richardson Bay, Bloody Falls (outside the park boundary) and near Heart Lake cemetery. Although previously treated as Ranunculus cymbalaria Pursh (Porsild & Cody, 1980; Whittemore, 1997), molecular evidence supports recognition of this taxon in the genus Halerpestes Greene (Hörandl et al., 2005; Paun et al., 2005; Emadzade et al., 2010). On mainland Nunavut recorded from Bathurst Inlet and a site to the west (Porsild & Cody, 1980; Whittemore, 1997; Bennett, 2015), Rankin Inlet (Korol, 1992) and Chesterfield Inlet (Cody, 1996a). Elsewhere in the Canadian Arctic known from northwestern Victoria Island, Tuktut Nogait National Park and vicinity, other Northwest Territories sites, and a few sites in northern Quebec and Labrador (Porsild & Cody, 1980; Aiken et al., 2007; Saarela et al., 2013a; Blondeau, 2015e).

Specimens Examined: Canada. Nunavut: Kitikmeot Region: Richardson Bay, confluence of Richardson and Rae rivers at Coronation Gulf, ca. 20 km WNW of Kugluktuk, 67°54′11.2″N, 115°32′27.4″W ± 200 m, 0 m, 8 July 2014, Saarela, Sokoloff & Bull 3669 (ALA, ALTA, CAN, O); clay slopes and beach on E side of Coppermine River, just above start of Bloody Falls, 67°44′9.4″N, 115°22′41.2″W ± 15 m, 40 m, 19 July 2014, Saarela, Sokoloff & Bull 4220 (CAN); gravel roadside SW of Kugluktuk, S side of road to Heart Lake cemetery, just beyond sewage retention pond, 5.59 km SW of mouth of Coppermine River, 67°48′39″N, 115°12′38.7″W ± 25 m, 46 m, 23 July 2014, Saarela, Sokoloff & Bull 4325 (CAN, UBC).

Ranunculus arcticus Richardson, Fig. S44—Birdfoot buttercup | Circumpolar–alpine

Recorded previously from Kugluktuk (Cody, 1954b; Porsild & Cody, 1980). We made collections at Coppermine Mountains, Expeditor Cove and Kugluk (Bloody Falls) Territorial Park. Elsewhere in the Canadian Arctic recorded from Baffin, Banks, Coats, Ellesmere, Melville, Southampton and Victoria islands, and across the mainland (Porsild & Cody, 1980; Cody, Scotter & Zoltai, 1989; Korol, 1992; Aiken et al., 2007; Saarela et al., 2013a; Blondeau, 2015e). This taxon was treated as Ranunculus pedatifidus var. affinis (R. B.) L. D. Benson by Whittemore (1997, who did not consider the name Ranunculus arcticus) and Ranunculus pedatifidus var. leiocarpus (Trautv.) Fernald by Cody (1954b). Elven et al. (2011) argued that Ranunculus arcticus is distinct from the Central Asian Ranunculus pedatifidus Sm.; we follow their taxonomy.

Specimens Examined: Canada. Nunavut: Kitikmeot Region: Coppermine [Kugluktuk], 67°49′36″N, 115°5′36″W, 4 July 1951, W. I. Findlay 85 (DAO-138206 01-01000620026); Coppermine [Kugluktuk], 67°49′36″N, 115°5′36″W, 23 June 1951, W. I. Findlay 32 (ALTA-VP-9117, DAO-138208 01-01000620025); Coronation Gulf, NW peninsula of Expeditor Cove, ca. 9.5 km NW of Kugluktuk, 67°52′40.4″N, 115°16′38.3″W ± 10 m, 15 m, 8 July 2014, Saarela, Sokoloff & Bull 3706 (CAN, UBC); flats atop and upper slopes of Coppermine Mountains, N/W side of Coppermine River, 67°14′43.7″N, 115°38′51.2″W ± 150 m, 422 m, 9 July 2014, Saarela, Sokoloff & Bull 3750 (ALA, CAN); Kugluk (Bloody Falls) Territorial Park, W side of Coppermine River, along ATV trail below slope of sand hill just below picnic bench/lookout area, 67°44′41.5″N, 115°22′14.9″W ± 15 m, 15 m, 17 July 2014, Saarela, Sokoloff & Bull 4153 (ALTA, CAN).

Ranunculus confervoides (Fr.) Fr.—Thread-leaved water buttercup | Amphi-Atlantic–European–Asian (N)–amphi-Beringian | Noteworthy Record

Newly recorded for the study area. Our collection from Kugluk (Bloody Falls) Territorial Park represents a minor range extension from a site west of the study area (67°32′50″N, 116°13′00″W, Reading 75, DAO; Cody & Reading, 2005). Elsewhere in the Canadian Arctic recorded from Baffin, Eglinton, Southampton and Victoria islands, and several sites across the mainland (Porsild & Cody, 1980; Aiken et al., 2007; Blondeau, 2015e). The water crowfoots (Ranunculus subgenus Batrachium (D. C.) A. Gray) are a taxonomically complex group that has been variously treated (reviewed in Elven et al., 2011; general discussion for the group is erroneously listed on the page for Ranunculus hyperboreus subsp. tricrenatus (Rupr.) Á. Löve & D. Löve, accessed 8 April 2015). Taxonomy follows Elven et al. (2011). In other floras, this taxon is treated as Ranunculus aquatilis var. eradicatus Laest. (Porsild & Cody, 1980) or Ranunculus aquatilis var. diffusus Withering (Whittemore, 1997; Aiken et al., 2007), the latter name treated as a synonym of Ranunculus trichophyllus Chaix in Elven et al. (2011). Ranunculus confervoides can be difficult to distinguish from Ranunculus subrigidus W. B. Drew (treated as Ranunculus aquatilis var. subrigidus (W. B. Drew) Breitung in Porsild & Cody (1980), and Whittemore (1997) treated them as a single taxon, Ranunculus aquatilis var. diffusus). The two differ primarily in the stiffness of their leaves (Porsild & Cody, 1980).

Specimens Examined: Canada. Nunavut: Kitikmeot Region: Kugluk (Bloody Falls) Territorial Park, rocky valley immediately SW of Bloody Falls, along rough marked section of Portage Trail, 67°44′34″N, 115°22′16″W ± 50 m, 20 m, 18 July 2014, Saarela, Sokoloff & Bull 4166 (ALA, ALTA, CAN, UBC).

Ranunculus gmelinii DC. subsp. gmelinii, Fig. 72A—Gmelin’s buttercup | European (NE)–Asian (N/C)–amphi-Beringian–North American (NW)

Figure 72 Ranunculus gmelinii subsp. gmelinii and Ranunculus hyperboreus.

Ranunculus gmelinii subsp. gmelinii: (A) habitat, Saarela et al. 3697. Ranunculus hyperboreus: (B) inflorescence, Saarela et al. 3696. (C) Habitat, Saarela et al. 3696. Photographs by J. M. Saarela (A) and R. D. Bull (B, C).

Recorded previously from Kugluktuk (Cody, 1954b; Porsild & Cody, 1980). We made collections of this white-flowered, aquatic buttercup at Expeditor Cove, on an island at the mouth of the Coppermine River and Kugluktuk. Elsewhere in the Canadian Arctic recorded from Banks, Prince Patrick, Melville and Victoria islands, and scattered sites across the mainland (Porsild & Cody, 1980; Korol, 1992; Cody, 1996a; Aiken et al., 2007; Saarela et al., 2013a; Blondeau, 2015e). Taxonomy follows Elven et al. (2011). The other infraspecific taxon, subsp. purshii (Richardson) Hultén, was recognised as a species (Ranunculus purshii Richardson) by Porsild & Cody (1980) and a synonym of Ranunculus gmelinii by Whittemore (1997); it is not known from the study area. Ranunculus gmelinii and Ranunculus hyperboreus are known to hybridise (Cayouette, Blondeau & Catling, 1997).

Specimens Examined: Canada. Nunavut: Kitikmeot Region: Coppermine [Kugluktuk], 67°49′36″N, 115°5′36″W, 6 August 1951, W. I. Findlay 255A (DAO-184406 01-01000620033); Coronation Gulf, NW peninsula of Expeditor Cove, ca. 9.6 km NW of Kugluktuk, 67°52′47.2″N, 115°16′40.3″W ± 3 m, 17 m, 8 July 2014, Saarela, Sokoloff & Bull 3697 (CAN); unnamed island just E (ca. 3.3 km) of Kugluktuk at mouth of Coppermine River, 67°49′29.2″N, 115°1′3.2″W ± 50 m, 1 m, 8 July 2014, Saarela, Sokoloff & Bull 3718 (CAN); flats below large overhanging cliffs above Coppermine River, just S of Kugluktuk, 67°48′56.7″N, 115°6′22.6″W ± 10 m, 2 m, 26 July 2014, Saarela, Sokoloff & Bull 4406 (ALA, CAN, UBC).

Ranunculus hyperboreus Rottb. subsp. hyperboreus, Figs. 72B and 72C—Far-northern buttercup | Circumpolar-alpine

Previously recorded from the study area (Porsild & Cody, 1980), but we were unable to locate a voucher specimen for confirmation. We made collections at Expeditor Cove and Kugluktuk. Widespread throughout the Canadian Arctic (Porsild & Cody, 1980; Cody, Scotter & Zoltai, 1984, 1989; Korol, 1992; Aiken et al., 2007; Saarela et al., 2013a; Blondeau, 2015e). Elven et al. (2011) recognise three subspecies, of which only subsp. hyperboreus is recorded for Canada.

Specimens Examined: Canada. Nunavut: Kitikmeot Region: Coronation Gulf, NW peninsula of Expeditor Cove, ca. 9.8 km NW of Kugluktuk, 67°52′44.7″N, 115°16′58.4″W ± 5 m, 18 m, 8 July 2014, Saarela, Sokoloff & Bull 3696 (CAN); W of Kugluktuk on tundra flats above Coppermine River, S of 1 Coronation Drive and N of power plant, 67°49′28.97″N, 115°5′0.2″W ± 100 m, 8 m, 25 July 2014, Saarela, Sokoloff & Bull 4364 (CAN).

Ranunculus nivalis L.—Snow buttercup | Circumpolar

Previously recorded from the study area (Porsild & Cody, 1980), but we were unable to locate a voucher specimen for confirmation. We made collections at Fockler Creek. Although uncommon in the study area, this species is widespread throughout the Canadian Arctic (Porsild & Cody, 1980; Cody, Reading & Line, 2003; Aiken et al., 2007; Saarela et al., 2013a; Blondeau, 2015e).

Specimens Examined: Canada. Nunavut: Kitikmeot Region: flats on W side of Fockler Creek, above spruce forest in creek valley, ca. 2.2 km S of Sandstone Rapids, Coppermine River, 67°25′49″N, 115°37′55″W ± 50 m, 152 m, 1 July 2014, Saarela, Sokoloff & Bull 3137 (CAN, UBC); NW-facing slope above tributary of Fockler Creek, ca. 2.4 km SSW of Sandstone Rapids, Coppermine River, 67°25′46″N, 115°38′49.4″W ± 50 m, 149 m, 3 July 2014, Saarela, Sokoloff & Bull 3294 (ALA, CAN).

Ranunculus pygmaeus Wahlenb.—Pygmy buttercup | Circumpolar–alpine

Previously recorded from Kugluktuk (Cody, 1954b; Porsild & Cody, 1980). We made two collections at Fockler Creek. Both are further inland than the range indicated on the map in Whittemore (1997) in the vicinity of the study area. Widespread throughout the Canadian Arctic (Porsild & Cody, 1980; Cody, Scotter & Zoltai, 1989; Cody, Reading & Line, 2003; Aiken et al., 2007; Saarela et al., 2013a; Blondeau, 2015e).

Specimens Examined: Canada. Nunavut: Kitikmeot Region: Coppermine [Kugluktuk], 67°49′36″N, 115°5′36″W, 22 July 1951, W. I. Findlay 167 (DAO-184514 01-01000620035); flats on W side of Fockler Creek, above spruce forest in creek valley, ca. 2.2 km S of Sandstone Rapids, Coppermine River, 67°25′49″N, 115°37′55″W ± 50 m, 152 m, 1 July 2014, Saarela, Sokoloff & Bull 3136 (CAN); NW-facing slope above tributary of Fockler Creek, ca. 2.4 km SSW of Sandstone Rapids, Coppermine River, 67°25′46″N, 115°38′49.4″W ± 50 m, 149 m, 3 July 2014, Saarela, Sokoloff & Bull 3293 (CAN, UBC).

Rosaceae [5/14]

Comarum palustre L.—Marsh cinquefoil | Circumboreal–polar | Noteworthy Record

Newly recorded for the study area. Our three collections close a distribution gap between Bathurst Inlet, Hood River, Great Bear Lake and Tuktut Nogait National Park and vicinity (Porsild & Cody, 1980; Gould & Walker, 1997; Saarela et al., 2013a). This prostrate shrub was growing in wet hummocky tundra around small ponds near Heart Lake with Andromeda polifolia, Betula glandulosa, Carex membranacea and Rubus chamaemorus; along the lower banks of a small stream running into Coronation Gulf; and along the edge of a wet sedge meadow with Carex aquatilis subsp. stans and Caltha palustris subsp. radicans near Kugluktuk. It occurs across most of the Nunavut mainland (Gussow, 1933; Porsild, 1950b; Porsild & Cody, 1980; Cody, Scotter & Zoltai, 1989; Korol, 1992; Cody & Reading, 2005; Bennett, 2015) and is recorded from Arctic sites in Northwest Territories and across northern Quebec and Labrador (Porsild & Cody, 1980; Saarela et al., 2013a; Bailleul, 2015). Taxonomy follows Elven et al. (2011) and Ertter & Reveal (2014a). Earlier treatments recognised it as Potentilla palustris (L.) Scop.

Specimens Examined: Canada. Nunavut: Kitikmeot Region: ca. 0.5 km SW of Heart Lake, SW of Kugluktuk, 7.5 km SW of mouth of Coppermine River, 67°47′52″N, 115°14′14.4″W ± 350 m, 66 m, 23 July 2014, Saarela, Sokoloff & Bull 4286 (CAN); sandy beach along Coronation Gulf, 3.9 km W of mouth of Coppermine River, 67°49′37.8″N, 115°10′31.8″W ± 50 m, 9 m, 23 July 2014, Saarela, Sokoloff & Bull 4338 (CAN); wetland at edge of extensive sandy floodplain of Coppermine River, below steep cliff above river and S of Kugluktuk, 67°48′56.9″N, 115°6′17″W ± 3 m, 2 m, 26 July 2014, Saarela, Sokoloff & Bull 4411 (CAN, UBC).

Dasiphora fruticosa (L.) Rydb., Fig. 73—Shrubby cinquefoil | Disjunct circumboreal

Figure 73 Dasiphora fruticosa.

(A) Habitat, Saarela et al. 3123. (B) Inflorescence, Saarela et al. 3123. Photographs by R. D. Bull.

Recorded previously from Bloody Falls and Kugluktuk (Cody, 1954b; Porsild & Cody, 1980) and noted as occurring “on the Coppermine River near the sea” by Richardson (1851). We made collections at Fockler Creek, Melville Creek, Kugluk (Bloody Falls) Territorial Park and Kugluktuk. This shrub species is common along the Coppermine River valley, extending to the Arctic coast. On mainland Nunavut recorded as far east as Bathurst Inlet (Porsild & Cody, 1980) and Hood River (Gould & Walker, 1997), and recorded across Arctic mainland Northwest Territories (Porsild & Cody, 1980; Saarela et al., 2013a). Not known from the Canadian Arctic Archipelago. This variable species (syn. Potentilla fruticosa L.) has been treated as one or more taxa (Klackenberg, 1983). Authors have recognised North American plants as a species, Dasiphora floribunda Pursh, distinct from the Old World Dasiphora fruticosa s.s. (Klackenberg, 1983); as a subspecies, Dasiphora fruticosa subsp. floribunda (Pursh) Kartesz (Cody, 2000; Elven et al., 2011); or as a single polymorphic species without further division (Ertter & Reveal, 2014b). We follow the latter treatment.

Specimens Examined: Canada. Nunavut: Kitikmeot Region: Bloody Falls, 67°44′N, 115°23′W, 18 July 1951, W. I. Findlay 133 (DAO-182684 01-01000619638); Coppermine [Kugluktuk], 67°49′36″N, 115°5′36″W, 30 July 1951, W. I. Findlay 213 (DAO-182683 01-01000619639); Kugluktuk, 67.821267°N, 115.08535°W, 22 July 2006, J. Davis 636 (CAN-597639); Coppermine River, Fort Hearne–Bloody Falls [67.7761972°N, 115.2037222°W ± 7.5 km], 1931, A. M. Berry 12 (CAN-71502); Coppermine [Kugluktuk], Coronation Gulf, at mouth of Coppermine River, back of the village [67.822146°N, 115.078387°W ± 0.5 km], 4 August 1948, H. T. Shacklette 3333 (CAN-200225); Coppermine [Kugluktuk] [67°49′36″N, 115°5′36″W ± 1.5 km], R. D. Wood s.n. (CAN-265510); Coppermine [Kugluktuk], vic. of hamlet and airstrip, 67.78°N, 115.5°W ± 3,615 m, 23 June 1999, C. L. Parker & I. Jonsdottir 9099 (ALA, as Pentaphylloides floribunda (Pursh) Á. Löve, not seen); flats on W side of Fockler Creek, above spruce forest in creek valley, ca. 2.2 km S of Sandstone Rapids, Coppermine River, 67°25′49″N, 115°37′55″W ± 50 m, 152 m, 1 July 2014, Saarela, Sokoloff & Bull 3123 (CAN, UBC); confluence of Coppermine River and Melville Creek, just W of Coppermine Mountains, 67°15′52″N, 115°30′55.3″W ± 350 m, 178–190 m, 7 July 2014, Saarela, Sokoloff & Bull 3492 (ALA, CAN); Kugluk (Bloody Falls) Territorial Park, rocky cliffs and ledges directly above (W side) of Bloody Falls, just S of heavily used day-use/fishing area, 67°44′40.1″N, 115°22′4.9″W ± 20 m, 8 m, 12 July 2014, Saarela, Sokoloff & Bull 3817 (ALTA, CAN); W of Kugluktuk on tundra flats above Coppermine River, S of 1 Coronation Drive and N of power plant, 67°49′28.97″N, 115°5′0.2″W ± 100 m, 8 m, 25 July 2014, Saarela, Sokoloff & Bull 4383 (CAN, O).

Dryas integrifolia Vahl subsp. integrifolia, Fig. S45—Mountain avens | Amphi-Beringian–North American (N)

Recorded previously from Bloody Falls and Kugluktuk (Cody, 1954b; Porsild & Cody, 1980). We made collections at Fockler Creek and Kugluk (Bloody Falls) Territorial Park. Widespread across the Canadian Arctic (Porsild & Cody, 1980; Cody, Scotter & Zoltai, 1989; Korol, 1992; Aiken et al., 2007; Saarela et al., 2013a; Bailleul, 2015). Taxonomy follows Elven et al. (2011) and Springer & Parfitt (2014), who recognise two subspecies. There is a collection of the northwestern North American subsp. sylvatica (Hultén) Hultén mapped for the study area in Porsild & Cody (1980), but we were unable to locate a voucher specimen and do not accept this record pending confirmation.

Specimens Examined: Canada. Nunavut: Kitikmeot Region: Bloody Falls on Coppermine River, 67°44′N, 115°23′W, 18 July 1951, W. I. Findlay 140 (DAO-183098 01-01000619641); Coppermine [Kugluktuk], 67°49′36″N, 115°5′36″W, 24 June 1951, W. I. Findlay 36 (DAO-183108 01-01000619642); Kugluktuk, upper tundra slope overlooking Coppermine River [67°49′36″N, 115°5′36″W], 14 June 2000, L. K. Benjamin s.n. (ACAD-ECS015867); Kugluktuk, overlooking Coppermine River [67°49′36″N, 115°5′36″W], 16 July 2000, L. K. Benjamin s.n. (ACAD-ECS015868); Kugluktuk, rocky slopes of North Hill, 67°49′29.6″N, 115°6′31″W ± 50 m, 50 m, 29 June 2014, Saarela, Sokoloff & Bull 3062 (ALA, ALTA, CAN); old riverbed of Fockler Creek, ca. 2.3 km SSE of Sandstone Rapids, Coppermine River, 67°25′45.7″N, 115°37′21.8″W ± 25 m, 166 m, 2 July 2014, Saarela, Sokoloff & Bull 3181 (CAN, UBC); Kugluk (Bloody Falls) Territorial Park, rocky cliffs and ledges directly above (W side) of Bloody Falls, just S of heavily used day-use/fishing area, 67°44′40.1″N, 115°22′4.9″W ± 20 m, 8 m, 12 July 2014, Saarela, Sokoloff & Bull 3832 (CAN).

Potentilla anserina subsp. groenlandica Tratt.—Greenland silverweed | Amphi-Beringian–North American (N)–amphi-Atlantic–European (N) | Noteworthy Record

This halophytic coastal taxon is newly recorded for the study area, and our single collection from Richardson Bay fills a distribution gap between sites at Paulatuk and the Lower Brock River lagoon (Saarela et al., 2013a) and Bathurst Inlet (Porsild & Cody, 1980; Bennett, 2015). Elsewhere in the Canadian Arctic recorded from Baffin and Victoria islands, the Mackenzie Delta area, along the western shore of Hudson Bay and numerous sites in northern Quebec and Labrador (Porsild & Cody, 1980; Korol, 1992; Aiken et al., 2007; Saarela et al., 2013a; Bailleul, 2015). It is treated as Potentilla egedii Wormsk. in Porsild & Cody (1980), a heterotypic synonym of the currently accepted name. Taxonomy follows Elven et al. (2011) and Elven & Murray (2014).

Specimens Examined: Canada. Nunavut: Kitikmeot Region: Richardson Bay, confluence of Richardson and Rae rivers at Coronation Gulf, ca. 20 km WNW of Kugluktuk, 67°54′11.2″N, 115°32′27.4″W ± 200 m, 0 m, 8 July 2014, Saarela, Sokoloff & Bull 3665 (CAN, UBC).

Potentilla arenosa (Turcz.) Juz. subsp. arenosa—Bluff cinquefoil | Asian (N/C)–amphi-Beringian–North American (N) | Noteworthy Record

Newly recorded for the study area. We made collections at Fockler Creek and Kugluk (Bloody Falls) Territorial Park. First collected in the study area in 1949, but the specimen from Kugluktuk (Porsild 17181) was determined by Porsild as Potentilla nivea subsp. chamissonis (Hultén) Hiitonen, and mapped under that name in Porsild & Cody (1980). This specimen has been re-determined to Potentilla arenosa subsp. arenosa (det. R. Elven, stet! J. M. Saarela). Findlay 86 is a mixed sheet with Potentilla nivea. Elsewhere in the Canadian Arctic recorded from Axel Heiberg, Banks, Devon, Melville and Victoria islands, and mainland sites west of the study area (a western Hudson Bay record in Porsild & Cody, 1980 needs confirmation) (Porsild & Cody, 1980; Aiken et al., 2007; Saarela et al., 2013a). The taxon is at the edge of its range in the study area. Taxonomy follows Elven, Murray & Ertter (2014b). Potentilla nivea subsp. chamissonis is a synonym of Potentilla arenosa subsp. chamissonis.

Specimens Examined: Canada. Nunavut: Kitikmeot Region: Coppermine [Kugluktuk], vicinity of post [67°49′36″N, 115°5′36″W ± 1.5 km], 26 July 1949, A. E. Porsild 17181 (CAN-127962); Coppermine [Kugluktuk], 67°49′36″N, 115°5′36″W, 4 July 1951, W. I. Findlay 86 (DAO-182828); old riverbed of Fockler Creek, 67°25′48″N, 115°37′33″W ± 25 m, 153 m, 1 July 2014, Saarela, Sokoloff & Bull 3147b (CAN); second ridge N of Fockler Creek, ca. 1.9 km SSE of Sandstone Rapids, Coppermine River, 67°26′2.4″N, 115°37′26.5″W ± 25 m, 187 m, 2 July 2014, Saarela, Sokoloff & Bull 3211 (CAN, UBC); E side of Fockler Creek, in valley just above creek’s confluence with the Coppermine River, ca. 1.4 km SSW of Sandstone Rapids, 67°26′21.4″N, 115°38′54″W ± 5 m, 140 m, 4 July 2014, Saarela, Sokoloff & Bull 3359b (CAN); Kugluk (Bloody Falls) Territorial Park, upper ledges of rocky (gabbro) S-facing cliffs above the start of Bloody Falls (W bank of River), just E of Portage Trail, 67°44′21.7″N, 115°22′42.2″W ± 25 m, 46 m, 14 July 2014, Saarela, Sokoloff & Bull 3940a (ALA, CAN); SSW-facing slopes above start of Bloody Falls, SE side of Coppermine River, 67°44′12.5″N, 115°22′31″W ± 50 m, 50–60 m, 19 July 2014, Saarela, Sokoloff & Bull 4210 (CAN, O).

Potentilla arenosa subsp. chamissonis (Hultén) Elven & D. F. Murray—Chamisso’s cinquefoil | North American (NE)–amphi-Atlantic–European (N)–Asian (NW)

A collection from the Kugluktuk area was determined and mapped (Porsild & Cody, 1980) under the name Potentilla nivea subsp. chamissonis, now a synonym of this taxon (Elven, Murray & Ertter, 2014b). We did not collect it in 2014. Elsewhere in the Canadian Arctic recorded from Baffin, Banks, Devon, Ellesmere, King William, Southampton and Victoria islands, and mainland sites (Porsild & Cody, 1980; Cody, Scotter & Zoltai, 1984; Aiken et al., 2007; Bailleul, 2015).

Specimens Examined: Canada. Nunavut: Kitikmeot Region: Coppermine [Kugluktuk] Dist., 1 August 1940, L. Ross 17 (CAN-531002).

Potentilla arenosa subsp. chamissonis × Potentilla nivea—Noteworthy Record

Plants with a combination of floccose and straight hairs on petioles are considered hybrids of Potentilla arenosa subsp. chamissonis and Potentilla nivea. These hybrid plants are newly recorded for the study area. We made collections at Fockler Creek and Kugluktuk. The name P. ×prostrata Rottb., based on plants from Greenland, has been used for such hybrids (Soják, 1989; Elven et al., 2011; Léveillé-Bourret et al., 2014; Bailleul, 2015). Elven, Murray & Ertter (2014b) did not include this hybrid in their key, as they did not consider it to be stabilised as a recognisable taxon. In their treatment, our collections key to Potentilla nivea, which they describe as having “long hairs…sometimes sparse to common (less so than cottony hairs)”. In all of the specimens determined here as putative hybrids, long hairs are less common than cottony ones.

Specimens Examined: Canada. Nunavut: Kitikmeot Region: Kugluktuk, rocky slopes of North Hill, 67°49′29.6″N, 115°6′31″W ± 50 m, 50 m, 29 June 2014, Saarela, Sokoloff & Bull 3043a (CAN, UBC); spruce forest along Fockler Creek, 67°25′45.7″N, 115°37′21.8″W ± 25 m, 166 m, 2 July 2014, Saarela, Sokoloff & Bull 3195a (CAN); E side of Fockler Creek, in valley just above creek’s confluence with the Coppermine River, ca. 1.4 km SSW of Sandstone Rapids, 67°26′21.4″N, 115°38′54″W ± 5 m, 140 m, 4 July 2014, Saarela, Sokoloff & Bull 3359a (CAN); W shore of Tundra Lake, ca. 4.3 km SE of Sandstone Rapids, Coppermine River, 67°25′43.7″N, 115°33′6.8″W ± 4 m, 262 m, 5 July 2014, Saarela, Sokoloff & Bull 3442a (CAN, O).

Potentilla biflora Willd. ex Schltdl., Fig. 74A—Two-flower cinquefoil | Asian (C) & amphi-Beringian | Noteworthy Record

Figure 74 Potentilla biflora and Potentilla hyparctica.

Potentilla biflora: (A) habit, Saarela et al. 3152. Potentilla hyparctica: (B) habit, Kugluktuk, Nunavut, 30 June 2014. Photographs by J. M. Saarela (A) and P. C. Sokoloff (B).

Newly recorded for the study area. We made collections in the Subarctic portion of the study area, at Fockler Creek and Big Creek. Previously known from the mainland Arctic of Alaska, to a site just east of Northwest Territories/Nunavut border (Porsild & Cody, 1980; Cody, Scotter & Zoltai, 1992; Saarela et al., 2013a), and four collections were recently reported from sites near the study area (Reading s.n., 2, 523, 525, DAO; Cody & Reading, 2005). A previously unpublished collection extends the range of the species considerably east to the Burnside River, southwest of Bathurst Inlet (Nunavut: Kitikmeot Region: Burnside River Ridge above Burnside River downstream from Kathawachaga Lake, near Nadlok Island, 66°16′N, 110°14′W, 13 July 1986, M. E. Gordon s.n., CAN-523650!). A collection mapped from the Cambridge Bay area on Victoria Island (Porsild & Cody, 1980; McJannet et al., 1993) and noted in subsequent publications (Cody & Reading, 2005; Elven, Murray & Ertter, 2014a) is a mapping error (Aiken et al., 2007). There are no confirmed records from the Canadian Arctic Archipelago.

Specimens Examined: Canada. Nunavut: Kitikmeot Region: old riverbed of Fockler Creek, ca. 2.3 km SSE of Sandstone Rapids, Coppermine River, 67°25′45.7″N, 115°37′21.8″W ± 25 m, 166 m, 1 July 2014, Saarela, Sokoloff & Bull 3152 (CAN, UBC); S of Fockler Creek, above small tributary of Fockler Creek, ca. 2.3 km S of Sandstone Rapids, Coppermine River, 67°25′46.3″N, 115°38′2.5″W ± 100 m, 156 m, 6 July 2014, Saarela, Sokoloff & Bull 3456 (CAN, O); forest and slopes at confluence of Big Creek and Coppermine River, N side of Coppermine River, S side of Coppermine Mountains, 67°14′29.3″N, 116°2′44.5″W ± 250 m, 180–199 m, 7 July 2014, Saarela, Sokoloff & Bull 3542 (ALA, CAN).

Potentilla hyparctica Malte subsp. hyparctica, Fig. 74B—Arctic cinquefoil | Circumpolar | Noteworthy Record

Our two collections from Fockler Creek are the first records for the study area and represent a southwestern range extension. At one site the plants were gathered in a meadow (old riverbed) below a north-facing slope on the edge of dense birch thicket adjacent to the spruce forest along Fockler Creek, growing with Castilleja caudata, Chamerion latifolium, Dasiphora fruticosa, Hedysarum americanum and Papaver hultenii. The other site was a nearby snowbed habitat, where plants grew with Anemone parviflora, Arctous rubra, Carex podocarpa, Cassiope tetragona subsp. tetragona, Dasiphora fruticosa and Salix reticulata. Both of our collections are from Subarctic locations, an ecozone in which this subspecies has apparently not been recorded previously. This subspecies is primarily distributed throughout the high Arctic and eastern Arctic islands and northeastern mainland Nunavut (Aiken et al., 2007), and is also recorded from a small area in northern Quebec (Bailleul, 2015). The nearest sites are on the Adelaide Peninsula and Banks Island (Aiken et al., 2007). The other subspecies, subsp. elatior (Abrom.) Elven & D. F. Murray, has a more southerly low Arctic distribution (Aiken et al., 2007; Murray & Elven, 2007; Elven et al., 2011; Saarela et al., 2013a; Elven et al., 2014). Aiken et al. (2007) mapped the infraspecific taxa as “northern race” and “southern race”. A collection identified as Potentilla hyparctica s.l. from the nearby Big Bend area of the Coppermine River (Reading 16, DAO—Cody, Reading & Line, 2003) is subsp. elatior, which is not recorded for the study area.

Specimens Examined: Canada. Nunavut: Kitikmeot Region: old riverbed of Fockler Creek, ca. 2.3 km SSE of Sandstone Rapids, Coppermine River, 67°25′45.7″N, 115°37′21.8″W ± 25 m, 166 m, 2 July 2014, Saarela, Sokoloff & Bull 3178 (CAN, UBC); NW-facing slope above tributary of Fockler Creek, ca. 2.4 km SSW of Sandstone Rapids, Coppermine River, 67°25′46″N, 115°38′49.4″W ± 50 m, 149 m, 3 July 2014, Saarela, Sokoloff & Bull 3296 (CAN).

Potentilla nivea L.—Snow cinquefoil | Circumpolar-alpine

Previously recorded from Kugluktuk (Porsild & Cody, 1980). We made collections at Fockler Creek, Expeditor Cove, Kugluk (Bloody Falls) Territorial Park and Kugluktuk. Elsewhere in the Canadian Arctic recorded from Baffin and Southampton islands, and across the mainland (Porsild & Cody, 1980; Cody, Reading & Line, 2003; Cody & Reading, 2005; Aiken et al., 2007; Bailleul, 2015). Taxonomy follows Elven, Murray & Ertter (2014b), which seems to match the circumscription of Potentilla nivea subsp. nivea in Porsild & Cody (1980). Findlay 86 is a mixed sheet with Potentilla arenosa subsp. arenosa.

Specimens Examined: Canada. Nunavut: Kitikmeot Region: bluff west of Coppermine [Kugluktuk] [67°49′36″N, 115°5′36″W ± 1.5 km], 7 July 1955, R. E. Miller 25 (CAN-241991); Coppermine [Kugluktuk], 67°49′36″N, 115°5′36″W, 4 July 1951, W. I. Findlay 86 (DAO-182828); old riverbed of Fockler Creek, 67°25′48″N, 115°37′33″W ± 25 m, 153 m, 1 July 2014, Saarela, Sokoloff & Bull 3147a (CAN); W shore of Tundra Lake, ca. 4.3 km SE of Sandstone Rapids, Coppermine River, 67°25′43.7″N, 115°33′6.8″W ± 4 m, 262 m, 5 July 2014, Saarela, Sokoloff & Bull 3442b (ALA, ALTA, CAN); Coronation Gulf, NW peninsula of Expeditor Cove, ca. 9.5 km NW of Kugluktuk, 67°52′39.5″N, 115°16′43.8″W ± 5 m, 14 m, 8 July 2014, Saarela, Sokoloff & Bull 3704 (CAN, UBC); Coronation Gulf, NW peninsula of Expeditor Cove, ca. 9.5 km NW of Kugluktuk, 67°52′39.5″N, 115°16′43.8″W ± 5 m, 14 m, 8 July 2014, Saarela, Sokoloff & Bull 3705 (CAN); Kugluk (Bloody Falls) Territorial Park, upper ledges of rocky (gabbro) S-facing cliffs above the start of Bloody Falls (W bank of River), just E of Portage Trail, 67°44′21.7″N, 115°22′42.2″W ± 25 m, 46 m, 14 July 2014, Saarela, Sokoloff & Bull 3940b (CAN, MO, O); rocky cliffs on S side of Kugluktuk, 67°49′13″N, 115°5′55.8″W ± 50 m, 65 m, 26 July 2014, Saarela, Sokoloff & Bull 4403 (CAN, O).

Potentilla pulchella R. Br.—Pretty cinquefoil, branching cinquefoil | Circumpolar | Noteworthy Record

Newly recorded for the study area. We made a single collection of this often-coastal species near the seashore along Richardson Bay. Elsewhere on mainland Nunavut known from the Boothia and Melville Peninsulas, the west coast of Hudson Bay and a site just east of the study area (Porsild & Cody, 1980). To the west recorded from Paulatuk and the Brock Lagoon (Saarela et al., 2013a) and other mainland sites (Porsild & Cody, 1980). Also recorded from northern Quebec (Bailleul, 2015) and widespread throughout the Canadian Arctic Archipelago (Aiken et al., 2007).

Specimens Examined: Canada. Nunavut: Kitikmeot Region: Richardson Bay, confluence of Richardson and Rae rivers at Coronation Gulf, ca. 20 km WNW of Kugluktuk, 67°54′11.2″N, 115°32′27.4″W ± 200 m, 0 m, 8 July 2014, Saarela, Sokoloff & Bull 3691 (CAN).

Potentilla tikhomirovii Juz.—Tikhomirov’s cinquefoil | Circumpolar? | Noteworthy Record

Newly recorded for the study area, but first collected in the area in 1951. The collection Findlay 12, gathered in Kugluktuk, was identified as Potentilla hookeriana Lehm. in Cody (1954b) but re-determined by us to this species. A nearby collection (Findlay 183) from the mouth of the Napaaktoktok River (just outside the study area) is a mixed sheet with Potentilla arenosa subsp. arenosa. We made collections at Fockler Creek, Kugluktuk and Kugluk (Bloody Falls) Territorial Park. Taxonomy follows Elven et al. (2011) and Elven, Murray & Ertter (2014b). This taxon is thought to be a hybrid of Potentilla arenosa subsp. arenosa and Potentilla hyparctica, and may have multiple origins (Elven, Murray & Ertter, 2014b). Until recently this name had not been used in the North American literature, and thus its Arctic distribution is unclear at present, pending revision of herbarium material.

Specimens Examined: Canada. Nunavut: Kitikmeot Region: Coppermine [Kugluktuk], 67°49′36″N, 115°5′36″W, 14 June 1951, W. I. Findlay 12 (DAO-182827 01-01000562765); Kugluktuk, rocky slopes of North Hill, 67°49′29.6″N, 115°6′31″W ± 50 m, 50 m, 29 June 2014, Saarela, Sokoloff & Bull 3043b (CAN); Kugluktuk, along Coronation Street roadside, 67°49′42″N, 115°6′6″W ± 50 m, 1 m, 30 June 2014, Saarela, Sokoloff & Bull 3105 (CAN, UBC); spruce forest along Fockler Creek, 67°25′45.7″N, 115°37′21.8″W ± 25 m, 166 m, 2 July 2014, Saarela, Sokoloff & Bull 3195b (CAN); Kugluk (Bloody Falls) Territorial Park, rocky valley immediately SW of Bloody Falls, along rough marked section of Portage Trail, 67°44′34″N, 115°22′16″W ± 50 m, 20 m, 13 July 2014, Saarela, Sokoloff & Bull 3893 (ALA, ALTA, CAN, O); Kugluk (Bloody Falls) Territorial Park, rocky valley immediately SW of Bloody Falls, along rough marked section of Portage Trail, 67°44′34″N, 115°22′16″W ± 50 m, 20 m, 18 July 2014, Saarela, Sokoloff & Bull 4165 (CAN); SE edge of Kugluktuk, rocky cliffs overlooking Coppermine River, 67°49′9.2″N, 115°5′40.4″W ± 50 m, 28 m, 24 July 2014, Saarela, Sokoloff & Bull 4355 (CAN); W of Kugluktuk on tundra flats above Coppermine River, S of 1 Coronation Drive and N of power plant, 67°49′28.97″N, 115°5′0.2″W ± 100 m, 8 m, 25 July 2014, Saarela, Sokoloff & Bull 4386 (CAN, O).

Rubus arcticus subsp. acaulis (Michx.) Focke, Fig. 75—Stemless raspberry | Amphi-Beringian (E)?–North American (N) | Noteworthy Record

Figure 75 Rubus arcticus subsp. acaulis.

(A) Inflorescence, Saarela et al. 3190. (B) Habitat, Saarela et al. 3190. (C) Habit, Saarela et al. 3190. Photographs by R. D. Bull (A, C) and J. M. Saarela (B).

Newly recorded for the study area, and a northeastern range extension from the nearest known sites along eastern Great Bear Lake (Porsild & Cody, 1980). It grew in small patches throughout the spruce forest along Fockler Creek, with Betula glandulosa, Juniperus communis subsp. depressa, Lupinus arcticus, Pyrola grandiflora, Shepherdia canadensis and Vaccinium vitis-idaea. Along nearby Sleigh Creek it was locally common in the understory of a dense willow forest (Salix alaxensis), growing with Carex podocarpa, Chamerion angustifolium and Dasiphora fruticosa. We encountered one fairly large population in Kugluk (Bloody Falls) Territorial Park, the first Arctic record for Nunavut, where plants grew in a fairly dense willow thicket with Arctous rubra, Betula glandulosa, Empetrum nigrum and Senecio lugens. Elsewhere in Nunavut recorded from the southeast mainland (Porsild, 1950b; Porsild & Cody, 1980; Cody, Reading & Line, 2003; Cody & Reading, 2005) and Akimiski Island (Blaney & Kotanen, 2001). It is distributed across boreal North America and reaches the Arctic in the Mackenzie Delta area, Northwest Territories, northern Yukon, and northern Quebec and Labrador (Porsild & Cody, 1980; Bailleul, 2015). This taxon has been variously recognised as a species, Rubus acaulis Michx. (Porsild & Cody, 1980; Elven et al., 2011), or a subspecies (Cody, 2000; Alice et al., 2014); we follow the latter approach.

Specimens Examined: Canada. Nunavut: Kitikmeot Region: spruce forest along Fockler Creek, ca. 2.3 km SSE of Sandstone Rapids, Coppermine River, 67°25′45.7″N, 115°37′21.8″W ± 25 m, 166 m, 2 July 2014, Saarela, Sokoloff & Bull 3190 (CAN, UBC); Sleigh Creek, just downstream (W) of Tundra Lake, ca. 3.7 km SE of Sandstone Rapids, Coppermine River, 67°26′2.2″N, 115°33′42.9″W ± 25 m, 229 m, 5 July 2014, Saarela, Sokoloff & Bull 3447 (ALA, CAN); confluence of Coppermine River and Melville Creek, just W of Coppermine Mountains, 67°15′52″N, 115°30′55.3″W ± 350 m, 178–190 m, 7 July 2014, Saarela, Sokoloff & Bull 3511 (ALTA, CAN); Kugluk (Bloody Falls) Territorial Park, SE-facing slope above small stream in deep gully that runs into Coppermine River just below Bloody Falls, ca. 1 km W of Bloody Falls, 67°44′41.2″N, 115°23′34.8″W ± 50 m, 49 m, 15 July 2014, Saarela, Sokoloff & Bull 4030 (CAN, O).

Rubus chamaemorus L.—Cloudberry, bakeapple | Circumboreal–polar

Recorded previously from Kugluktuk (Cody, 1954b; Porsild & Cody, 1980). We made collections at Fockler Creek, Kugluk (Bloody Falls) Territorial Park and near Heart Lake. Elsewhere in the Arctic recorded from Baffin, Coats, King William, Southampton and Victoria islands, and across the mainland (Porsild & Cody, 1980; Korol, 1992; Aiken et al., 2007; Saarela et al., 2013a; Bailleul, 2015; Bennett, 2015).

Specimens Examined: Canada. Nunavut: Kitikmeot Region: Coppermine [Kugluktuk], 67°49′36″N, 115°5′36″W, 17 July 1951, W. I. Findlay 128 (DAO-182420 01-01000619632); Coppermine [Kugluktuk], 67°49′36″N, 115°5′36″W, 10 July 1951, W. I. Findlay 109 (DAO-182421 01-01000619633); Kugluktuk, 67.814333°N, 115.19845°W, 23 July 2006, J. Davis 639 (CAN-597646); flats on W side of Fockler Creek, above spruce forest in creek valley, ca. 2.2 km S of Sandstone Rapids, Coppermine River, 67°25′49″N, 115°37′55″W ± 50 m, 152 m, 1 July 2014, Saarela, Sokoloff & Bull 3134 (CAN, UBC); Kugluk (Bloody Falls) Territorial Park, day-use area above Bloody Falls (at outhouse and fire pit), 67°44′36.8″N, 115°22′11.1″W ± 25 m, 28 m, 12 July 2014, Saarela, Sokoloff & Bull 3836 (ALA, CAN); ca. 0.5 km SW of Heart Lake, SW of Kugluktuk, 7.5 km SW of mouth of Coppermine River, 67°47′52″N, 115°14′14.4″W ± 350 m, 66 m, 23 July 2014, Saarela, Sokoloff & Bull 4281 (CAN).

Salicaceae [2/17]

Populus balsamifera L., Fig. 76—Balsam poplar | Amphi-Beringian (E)–North American

Figure 76 Populus balsamifera.

(A) Catkins, Saarela et al. 3457. (B) Habit, Saarela et al. 3457. (C) Habitat, Saarela et al. 3457. Photographs by R. D. Bull.

The distribution of this species at its northern range is summarised by Saarela et al. (2012), who reported Arctic stands in Tuktut Nogait National Park, Northwest Territories. In Nunavut, extralimital balsam poplar is known from a single collection made during the Canadian Arctic Expedition, 1913–18, in the study area (Johansen, 1924). A few small twigs from “trees … about ten feet [ca. 3 m] high” with “trunks [that] attained the thickness of a finger” growing in a deep gully just above Escape Rapid in the Coppermine River valley were collected by R. M. Anderson (Anderson 756, CAN) in winter (February 1916) when “all but the upper twigs were hidden by snow…” Johansen (1924) This depauperate collection was originally reported as Populus tremuloides Michx. (Macoun & Holm, 1921; Johansen, 1924), though Holm (1922) also considered it could be Populus balsamifera. Porsild & Cody (1980) reported the collection as Populus balsamifera, a determination that is likely correct (G. W. Argus, 2011, personal communication, 2011). We explored the slopes above Escape Rapids on the west side of the Coppermine River, but did not encounter this species or the deep gully mentioned. The gully may be on the east side of the river, or beyond the area we explored. We did, however, encounter one small but healthy stand of balsam poplar at a different site at Fockler Creek. The species was growing on a sparsely vegetated, ca. south-facing slope permeated with ground-nesting wasp holes, associated with Arctous rubra, Betula glandulosa, Dasiphora fruticosa, Juniperus communis subsp. depressa, Picea glauca and Saxifraga tricuspidata. One ramet about 6 ft. tall had fruits, while all other ramets were sterile and many small suckers were 10–40 cm tall. The species is rare in the study area. Elsewhere in the Canadian Arctic there are records from northern Labrador (Argus, 2015).

Specimens Examined: Canada. Nunavut: Kitikmeot Region: Gully Creek just above Escape Rapids, east side of Coppermine River [67.6167°N, 115.4833°W ± 1,000 m], February 1916, R. M. Anderson 756 (CAN-40681); S of Fockler Creek, S-facing slope on N side of small tributary of Fockler Creek, ca. 2.3 km S of Sandstone Rapids, Coppermine River, 67°25′46.3″N, 115°38′2.5″W ± 5 m, 156 m, 6 July 2014, Saarela, Sokoloff & Bull 3457 (ALA, ALTA, CAN, UBC, US).

Salix alaxensis (Andersson) Coville var. alaxensis, Fig. 77A—Felt-leaf willow | Asian (N)–amphi-Beringian–North American (NW)

Figure 77 Salix alaxensis var. alaxensis and Salix niphoclada.

Salix alaxensis var. alaxensis: (A) habit, Saarela et al. 4191. Salix niphoclada: (B) staminate catkins, Saarela et al. 4019. (C) Habit, Saarela et al. 4019. Photographs by J. M. Saarela (A), R. D. Bull (B), and P. C. Sokoloff (C).

Previously recorded from Kugluktuk (Porsild & Cody, 1980). We made collections at Fockler Creek, Bigtree River and Kugluk (Bloody Falls) Territorial Park. Taxonomy follows Argus (2010) and Elven et al. (2011), who recognise two varieties in the species. The other one, var. longistylis (Rydb.) C. K. Schneid., is not known from Nunavut. Porsild & Cody (1980) did not recognise infraspecific taxa. This willow is common in Bloody Falls/Kugluk Territorial Park, where it often grows as a large, upright shrub or small tree in the deep gullies of the sand hills. It is curious that Findlay did not collect this conspicuous species at Bloody Falls. Elsewhere in the Canadian Arctic recorded from Banks, Southampton and Victoria islands, mainland Nunavut and Northwest Territories, and northern Quebec (Porsild & Cody, 1980; Cody, Scotter & Zoltai, 1989; Korol, 1992; Cody & Reading, 2005; Aiken et al., 2007; Saarela et al., 2013a; Argus, 2015; Bennett, 2015).

Specimens Examined: Canada. Nunavut: Kitikmeot Region: Coppermine [Kugluktuk] [67°49′36″N, 115°5′36″W ± 1.5 km], 1962, J. A. Larsen s.n. (CAN-349130); old riverbed of Fockler Creek, ca. 2.3 km SSE of Sandstone Rapids, Coppermine River, 67°25′48″N, 115°37′33″W ± 25 m, 153 m, 1 July 2014, Saarela, Sokoloff & Bull 3149 (CAN, UBC); confluence of Coppermine and Bigtree rivers, 66°56′23.8″N, 116°21′3.2″W ± 100 m, 265 m, 7 July 2014, Saarela, Sokoloff & Bull 3599 (ALA, CAN); confluence of Coppermine and Bigtree rivers, 66°56′23.8″N, 116°21′3.2″W ± 100 m, 265 m, 7 July 2014, Saarela, Sokoloff & Bull 3600 (CAN); Kugluk (Bloody Falls) Territorial Park, rocky cliffs and ledges directly above (W side) of Bloody Falls, just S of heavily used day-use/fishing area, 67°44′40.1″N, 115°22′4.9″W ± 20 m, 8 m, 12 July 2014, Saarela, Sokoloff & Bull 3827 (CAN, MO, MT); SW-facing slopes of shallow gully in sand hills above Bloody Falls, SE side of Coppermine River across river from Kugluk (Bloody Falls) Territorial Park, 67°44′28.2″N, 115°22′3″W ± 15 m, 78 m, 19 July 2014, Saarela, Sokoloff & Bull 4191 (CAN, O).

Salix arbusculoides Andersson—Little-tree willow | North American (NW) | Noteworthy Record

This primarily Subarctic species is newly recorded for the study area. Our collection was made along Sleigh Creek, an area just outside the range for the taxon given in Argus (2007), from a tree ca. 20 ft. tall that grew with Carex podocarpa, Chamerion angustifolium, Dasiphora fruticosa, Rubus arcticus subsp. acaulis and Salix alaxensis var. alaxensis. The collection fills in a distribution gap between Bathurst Inlet, Hood River, eastern Great Bear Lake (Porsild & Cody, 1980; Gould & Walker, 1997) and just outside Tuktut Nogait National Park (Saarela et al., 2013a). Elsewhere in the Canadian Arctic recorded from a few mainland Nunavut sites near the treeline (Porsild & Cody, 1980).

Specimens Examined: Canada. Nunavut: Kitikmeot Region: Sleigh Creek, just downstream (W) of Tundra Lake, ca. 3.7 km SE of Sandstone Rapids, Coppermine River, 67°26′2.2″N, 115°33′42.9″W ± 25 m, 229 m, 5 July 2014, Saarela, Sokoloff & Bull 3448 (CAN, UBC).

Salix arctica Pall.—Arctic willow | Circumpolar–alpine

Previously recorded from Kugluktuk (Macoun & Holm, 1921, as Salix anglorum Cham.; Cody, 1954b; Porsild & Cody, 1980). We made collections at Kugluktuk, Fockler Creek and Kugluk (Bloody Falls) Territorial Park. Widespread throughout the Canadian Arctic (Porsild & Cody, 1980; Cody, Scotter & Zoltai, 1989; Korol, 1992; Cody & Reading, 2005; Aiken et al., 2007; Saarela et al., 2013a; Argus, 2015).

Specimens Examined: Canada. Nunavut: Kitikmeot Region: Coppermine [Kugluktuk] 67°49′36″N, 115°5′36″W, 20 June 1951, W. I. Findlay 24 (ACAD-30939, DAO-199179 01-01000677513); Coppermine [Kugluktuk], 67°49′36″N, 115°5′36″W, 20 June 1951, W. I. Findlay 25 (DAO-199178 01-01000677978); Coppermine [Kugluktuk], 67°49′36″N, 115°5′36″W, 15 June 1951, W. I. Findlay 11 (DAO-199214 01-01000677962); Kugluktuk, Bloody Falls Cabin [67.843056°N, 115.097222°W], 30 July 1962, J. A. Larsen s.n. (CAN-349632, det. G. W. Argus 1963); Coppermine [Kugluktuk] [67.8333°N, 115.1°W], 29 July 1962, J. A. Larsen 8 (CAN-349630, det. G. W. Argus 1963), s.n. (CAN-349626, det. G. W. Argus 1963); Coppermine [Kugluktuk], vicinity of post [67.8333°N, 115.1°W], 26 July 1949, A. E. Porsild 17167 (CAN-127633); Coppermine [Kugluktuk], in settlement near R.C. Mission [67.8333°N, 115.1°W], 4 July 1958, R. D. Wood (CAN-265586); Kugluktuk, rocky slopes of North Hill, 67°49′29.6″N, 115°6′31″W ± 50 m, 50 m, 29 June 2014, Saarela, Sokoloff & Bull 3050 (ALA, CAN); NW-facing slope above tributary of Fockler Creek, ca. 2.4 km SSW of Sandstone Rapids, Coppermine River, 67°25′46″N, 115°38′49.4″W ± 50 m, 149 m, 3 July 2014, Saarela, Sokoloff & Bull 3281 (CAN); NW-facing slope above tributary of Fockler Creek, ca. 2.4 km SSW of Sandstone Rapids, Coppermine River, 67°25′46″N, 115°38′49.4″W ± 50 m, 149 m, 3 July 2014, Saarela, Sokoloff & Bull 3282 (CAN); meadow just S of Tundra Lake, ca. 4.2 km SE of Sandstone Rapids, Coppermine River, 67°25′29.5″N, 115°33′50.4″W ± 50 m, 266 m, 5 July 2014, Saarela, Sokoloff & Bull 3432a (CAN); Kugluk (Bloody Falls) Territorial Park, slope above Bloody Falls (W side) just below Coppermine River narrows to Falls, between Portage Trail and river, 67°44′25.4″N, 115°22′31.2″W ± 25 m, 29 m, 12 July 2014, Saarela, Sokoloff & Bull 3852 (CAN, MT, US); Kugluk (Bloody Falls) Territorial Park, N-facing slopes of large mountain just S of start of Bloody Falls, W side of Coppermine River, 67°44′7.7″N, 115°23′30.4″W ± 15 m, 90 m, 14 July 2014, Saarela, Sokoloff & Bull 4008 (ALTA, CAN); Kugluk (Bloody Falls) Territorial Park, flats above boardwalk W of Bloody Falls, 67°44′34.5″N, 115°22′27″W ± 100 m, 135 m, 16 July 2014, Saarela, Sokoloff & Bull 4053 (CAN, QFA, WIN); Kugluk (Bloody Falls) Territorial Park, W side of Coppermine River, between Sandy Hills and Bloody Falls, 67°45′13.2″N, 115°22′6.3″W ± 3 m, 21 m, 17 July 2014, Saarela, Sokoloff & Bull 4148 (CAN, O); clay slopes and beach on E side of Coppermine River, just above start of Bloody Falls, 67°44′9.4″N, 115°22′41.2″W ± 15 m, 40 m, 19 July 2014, Saarela, Sokoloff & Bull 4214 (CAN, MO); SE edge of Kugluktuk, rocky cliffs overlooking Coppermine River, 67°49′9.2″N, 115°5′40.4″W ± 50 m, 28 m, 24 July 2014, Saarela, Sokoloff & Bull 4351 (CAN, K, NY).

Salix arctophila Cockerell ex A. Heller—Northern willow | North American (N)

Previously recorded from Kugluktuk (Porsild & Cody, 1980). We did not collect this species in 2014, but it was recently collected from the nearby Big Bend area of the Coppermine River (Reading 39-2, DAO). Elsewhere in the Canadian Arctic recorded from Baffin, Southampton and Victoria islands, and across the mainland (Porsild & Cody, 1980; Korol, 1992; Cody & Reading, 2005; Aiken et al., 2007; Saarela et al., 2013a; Argus, 2015).

Specimens Examined: Canada. Nunavut: Kitikmeot Region: Coppermine [Kugluktuk], vicinity of post [67°49′36″N, 115°5′36″W ± 1.5 km], 26 July 1949, A. E. Porsild 17168 (CAN-127634); Coppermine [Kugluktuk], vicinity of post [67°49′36″N, 115°5′36″W ± 1.5 km], 26 July 1949, A. E. Porsild 17166 (CAN-127635); Coppermine [Kugluktuk], 67°49′36″N, 115°5′36″W, 20 June 1951, W. I. Findlay 23 (DAO-199215 01-01000677979); Coppermine [Kugluktuk], 67°49′36″N, 115°5′36″W, 2 August 1951, W. I. Findlay 238 (DAO-199203 01-01000677515); Coppermine [Kugluktuk] [67°49′36″N, 115°5′36″W ± 1.5 km], 8 October 1962, J. A. Larsen 5 (CAN-349739).

Salix arctica × Salix arctophila

Two collections previously identified as Salix arctica (Findlay 10, 22; Cody, 1954b) are re-determined as this hybrid, which is newly recorded for the study area. The specimens combine morphological characteristics of both parental species, as described in Argus (2010).

Specimens Examined: Canada. Nunavut: Kitikmeot Region: Coppermine [Kugluktuk], 67°49′36″N, 115°5′36″W, 15 June 1951, W. I. Findlay 11 (DAO-199214 01-01000677962); Coppermine [Kugluktuk], 67°49′36″N, 115°5′36″W, 20 June 1951, W. I. Findlay 22 (DAO-199207 01-01000677963).

Salix glauca var. cordifolia (Pursh) Dorn—Beautiful willow | North American (NE) | Noteworthy Record

Previously recorded from Bloody Falls and Kugluktuk (Cody, 1954b; Porsild & Cody, 1980). This is a common species in the study area, and we made numerous collections at Fockler Creek, Melville Creek, Big Creek and Kugluk (Bloody Falls) Territorial Park. Porsild & Cody (1980) treated this taxon as Salix cordifolia var. callicarpea (Trautv.) Fernald and recorded it as occurring as far west on the mainland as Bathurst Inlet. Argus (1965) treated it as the “eastern phase” of Salix glauca and mapped it as far east as central former Keewatin district. Based on these maps, all collections from the study area represent a western range extension for the variety. Elven et al. (2011) recognised it as Salix glauca subsp. callicarpaea (Trautv.) Böcher. We follow Argus (2010).

Specimens Examined: Canada. Nunavut: Kitikmeot Region: Bloody Falls, 67°44′N, 115°23′W, 18 July 1951, W. I. Findlay 150 (DAO-5643 01-01000677969); Bloody Falls, 67°44′N, 115°23′W, 18 July 1951, W. I. Findlay 151 (DAO-5644 01-01000677505); Coppermine [Kugluktuk], 67°49′36″N, 115°5′36″W, W. I. Findlay 218 (DAO-5642 01-01000684838); Kugluktuk, 67.798017°N, 115.23075°W, 22 June 2006, J. Davis 606 (CAN-597643); Kugluktuk, airport, 21 July 2013, 67.81749°N, 115.13449°W, B. A. Bennett 13-0331 (BABY, det. B. A. Bennett, July 2013); flats on W side of Fockler Creek, above spruce forest in creek valley, ca. 2.2 km S of Sandstone Rapids, Coppermine River, 67°25′49″N, 115°37′55″W ± 50 m, 152 m, 1 July 2014, Saarela, Sokoloff & Bull 3130 (CAN, MO, MT, O); flats on W side of Fockler Creek, above spruce forest in creek valley, ca. 2.2 km S of Sandstone Rapids, Coppermine River, 67°25′49″N, 115°37′55″W ± 50 m, 152 m, 1 July 2014, Saarela, Sokoloff & Bull 3131 (ALTA, CAN, O); spruce forest along Fockler Creek, ca. 2.3 km SSE of Sandstone Rapids, Coppermine River, 67°25′45.7″N, 115°37′21.8″W ± 25 m, 166 m, 2 July 2014, Saarela, Sokoloff & Bull 3202 (CAN); S of Fockler Creek, S-facing slope on N side of small tributary flowing into Fockler Creek, ca. 2.3 km S of Sandstone Rapids, Coppermine River, 67°25′46.3″N, 115°38′2.5″W ± 25 m, 156 m, 3 July 2014, Saarela, Sokoloff & Bull 3237 (CAN, MO, MT); E side of Fockler Creek, just above its confluence with Coppermine River, ca. 1.1 km SW of Sandstone Rapids, 67°26′30.6″N, 115°39′4.3″W ± 50 m, 135 m, 4 July 2014, Saarela, Sokoloff & Bull 3373 (CAN, UBC); E side of Fockler Creek, just above its confluence with Coppermine River, ca. 1.1 km SW of Sandstone Rapids, 67°26′30.6″N, 115°39′4.3″W ± 50 m, 135 m, 4 July 2014, Saarela, Sokoloff & Bull 3374 (ALA, CAN); E side of Fockler Creek, just above its confluence with Coppermine River, ca. 1.1 km SW of Sandstone Rapids, 67°26′30.6″N, 115°39′4.3″W ± 50 m, 135 m, 4 July 2014, Saarela, Sokoloff & Bull 3375 (ALTA, CAN); confluence of Coppermine River and Melville Creek, just W of Coppermine Mountains, 67°15′52″N, 115°30′55.3″W ± 350 m, 178–190 m, 7 July 2014, Saarela, Sokoloff & Bull 3499 (CAN, O); confluence of Coppermine River and Melville Creek, just W of Coppermine Mountains, 67°15′52″N, 115°30′55.3″W ± 350 m, 178–190 m, 7 July 2014, Saarela, Sokoloff & Bull 3509 (CAN, UBC, US); confluence of Coppermine River and Melville Creek, just W of Coppermine Mountains, 67°15′52″N, 115°30′55.3″W ± 350 m, 178–190 m, 7 July 2014, Saarela, Sokoloff & Bull 3516 (CAN, MO); confluence of Coppermine River and Melville Creek, just W of Coppermine Mountains, 67°15′52″N, 115°30′55.3″W ± 350 m, 178–190 m, 7 July 2014, Saarela, Sokoloff & Bull 3523 (CAN, MT); forest and slopes at confluence of Big Creek and Coppermine River, N side of Coppermine River, S side of Coppermine Mountains, 67°14′29.3″N, 116°2′44.5″W ± 250 m, 180–199 m, 7 July 2014, Saarela, Sokoloff & Bull 3552 (CAN, UBC); S-facing slopes above Coppermine River, ca. 7.8 km NNE of Sandstone Rapids, 67°31′16.2″N, 115°36′52.1″W ± 50 m, 110 m, 8 July 2014, Saarela, Sokoloff & Bull 3641 (CAN); Kugluk (Bloody Falls) Territorial Park, rocky cliffs and ledges directly above (W side) of Bloody Falls, just S of heavily used day-use/fishing area, 67°44′40.1″N, 115°22′4.9″W ± 20 m, 8 m, 12 July 2014, Saarela, Sokoloff & Bull 3829 (CAN, WIN); Kugluk (Bloody Falls) Territorial Park, rocky cliffs and ledges directly above (W side) of Bloody Falls, just S of heavily used day-use/fishing area, 67°44′40.1″N, 115°22′4.9″W ± 20 m, 8 m, 12 July 2014, Saarela, Sokoloff & Bull 3830 (CAN, QFA, WIN); Kugluk (Bloody Falls) Territorial Park, day-use area above Bloody Falls (at outhouse and fire pit), 67°44′36.8″N, 115°22′11.1″W ± 25 m, 28 m, 12 July 2014, Saarela, Sokoloff & Bull 3841 (CAN, UBC, US, WIN); Kugluk (Bloody Falls) Territorial Park, day-use area above Bloody Falls (at outhouse and fire pit), 67°44′36.8″N, 115°22′11.1″W ± 25 m, 28 m, 12 July 2014, Saarela, Sokoloff & Bull 3842 (CAN, QFA); Kugluk (Bloody Falls) Territorial Park, slope above Bloody Falls (W side) just below Coppermine River narrows to Falls, between Portage Trail and river, 67°44′25.4″N, 115°22′31.2″W ± 25 m, 29 m, 12 July 2014, Saarela, Sokoloff & Bull 3850 (CAN, K, NY); Kugluk (Bloody Falls) Territorial Park, slope above Bloody Falls (W side) just below Coppermine River narrows to Falls, between Portage Trail and river, 67°44′25.4″N, 115°22′31.2″W ± 25 m, 29 m, 12 July 2014, Saarela, Sokoloff & Bull 3853 (ALA, ALTA, CAN); Kugluk (Bloody Falls) Territorial Park, gentle stream in shallow valley running into Coppermine River just W of Bloody Falls, 67°44′36.6″N, 115°22′59.3″W ± 20 m, 41 m, 15 July 2014, Saarela, Sokoloff & Bull 4012 (CAN, NY); Kugluk (Bloody Falls) Territorial Park, S-facing cliff (gabbro sill) above start of Bloody Falls, W side of Coppermine River, W side of Portage Trail, 67°44′23.2″N, 115°22′54.5″W ± 50 m, 57 m, 16 July 2014, Saarela, Sokoloff & Bull 4070 (CAN, K); Kugluk (Bloody Falls) Territorial Park, S-facing cliff (gabbro sill) above start of Bloody Falls, W side of Coppermine River, W side of Portage Trail, 67°44′23.2″N, 115°22′54.5″W ± 50 m, 57 m, 16 July 2014, Saarela, Sokoloff & Bull 4072 (ALA, CAN); SW-facing slope above Bloody Falls, SE side of Coppermine River, across river from Kugluk (Bloody Falls) Territorial Park, 67°44′27.2″N, 115°22′58″W ± 5 m, 68 m, 19 July 2014, Saarela, Sokoloff & Bull 4195 (CAN); SW-facing slope above Bloody Falls, SE side of Coppermine River, 67°44′27.2″N, 115°22′58″W ± 5 m, 68 m, 19 July 2014, Saarela, Sokoloff & Bull 4198 (CAN).

Salix glauca L. var. glauca—Grey-leaved willow | Amphi-Atlantic (?)–European (N) | Noteworthy Record

Several collections of Salix glauca L. were reported from Kugluktuk and Bloody Falls in Cody (1954b); no infraspecific taxa were recognised there. Most collections of Salix glauca in the study area fall under var. cordifolia, but two from Kugluk (Bloody Falls) Territorial Park better fit var. glauca, which is newly recorded for the study area.

Specimens Examined: Canada. Nunavut: Kitikmeot Region: Kugluk (Bloody Falls) Territorial Park, W side of Coppermine River, just above Bloody Falls, 67°44′22.6″N, 115°22′52″W ± 20 m, 40 m, 16 July 2014, Saarela, Sokoloff & Bull 4108 (CAN); SW-facing slope above Bloody Falls, SE side of Coppermine River, 67°44′27.2″N, 115°22′58″W ± 5 m, 68 m, 19 July 2014, Saarela, Sokoloff & Bull 4196 (CAN, UBC).

Salix niphoclada Rydb., Figs. 77B and 77C—Barren-ground willow | Amphi-Beringian (E)

Previously recorded from south of Kugluktuk (Porsild & Cody, 1980). Three collections determined as Salix glauca in Cody (1954b) are this species. We made collections at Fockler Creek, along the Coppermine River, Richardson Bay and Kugluk (Bloody Falls) Territorial Park. Elsewhere in the Canadian Arctic recorded from Banks and Victoria islands, mainland Northwest Territories and a few mainland Nunavut sites (Porsild & Cody, 1980; Aiken et al., 2007; Argus, 2007; Saarela et al., 2013a).

Specimens Examined: Canada. Nunavut: Kitikmeot Region: Coppermine [Kugluktuk], 67°49′36″N, 115°5′36″W, W. I. Findlay 65 (DAO-5641 01-01000684852); Coppermine [Kugluktuk], Cemetery Island, 67°50′N, 115°7′W, W. I. Findlay 121 (DAO-5646 01-01000684851); Coppermine [Kugluktuk], Cemetery Island, 67°50′N, 115°7′W, W. I. Findlay 122 (DAO-5645 01-01000684850); Coppermine [Kugluktuk], west bank of Coppermine River, about six miles from mouth [67.7668194°N, 115.252225°W ± 1 km], 4 July 1958, R. D. Wood s.n. (CAN-265585); E side of Fockler Creek, ridge above creek valley before its confluence with Coppermine River, ca. 1.8 km S of Sandstone Rapids, 67°26′3.9″N, 115°38′20.4″W ± 25 m, 168 m, 4 July 2014, Saarela, Sokoloff & Bull 3340 (CAN); E side of Fockler Creek, ridge above creek valley before its confluence with Coppermine River, ca. 1.8 km S of Sandstone Rapids, 67°26′3.9″N, 115°38′20.4″W ± 25 m, 168 m, 4 July 2014, Saarela, Sokoloff & Bull 3341 (CAN, QFA, WIN); E side of Fockler Creek, in valley just above creek’s confluence with the Coppermine River, ca. 1.4 km SSW of Sandstone Rapids, 67°26′21.4″N, 115°38′54″W ± 5 m, 140 m, 4 July 2014, Saarela, Sokoloff & Bull 3362 (CAN); E side of Fockler Creek, just above its confluence with Coppermine River, ca. 1.1 km SW of Sandstone Rapids, 67°26′30.6″N, 115°39′4.3″W ± 50 m, 135 m, 4 July 2014, Saarela, Sokoloff & Bull 3372 (ALTA, CAN); slopes on E side of Coppermine River, N of its confluence with Fockler Creek, ca. 0.8 km SW of Sandstone Rapids, 67°26′36.9″N, 115°38′50.1″W ± 50 m, 128 m, 4 July 2014, Saarela, Sokoloff & Bull 3387 (CAN, O); Coppermine River, sandstone cliffs above Sandstone Rapids, 67°27′29.6″N, 115°37′59.3″W ± 100 m, 110 m, 6 July 2014, Saarela, Sokoloff & Bull 3470 (CAN, MO); Coppermine River, sandstone cliffs above Sandstone Rapids, 67°27′29.6″N, 115°37′59.3″W ± 100 m, 110 m, 6 July 2014, Saarela, Sokoloff & Bull 3472 (CAN); S-facing slopes on W side of Coppermine River, about halfway between Escape Rapids and Muskox Rapids, 67°31′18.2″N, 115°36′20.1″W ± 150 m, 115 m, 8 July 2014, Saarela, Sokoloff & Bull 3626 (CAN); S-facing slopes on W side of Coppermine River, about halfway between Escape Rapids and Muskox Rapids, 67°31′18.2″N, 115°36′20.1″W ± 150 m, 115 m, 8 July 2014, Saarela, Sokoloff & Bull 3627 (CAN, MT); Richardson Bay, confluence of Richardson and Rae rivers at Coronation Gulf, ca. 20 km WNW of Kugluktuk, 67°54′11.2″N, 115°32′27.4″W ± 200 m, 0 m, 8 July 2014, Saarela, Sokoloff & Bull 3682 (CAN); Richardson Bay, confluence of Richardson and Rae rivers at Coronation Gulf, ca. 20 km WNW of Kugluktuk, 67°54′11.2″N, 115°32′27.4″W ± 200 m, 0 m, 8 July 2014, Saarela, Sokoloff & Bull 3683 (CAN); Richardson Bay, confluence of Richardson and Rae rivers at Coronation Gulf, ca. 20 km WNW of Kugluktuk, 67°54′11.2″N, 115°32′27.4″W ± 200 m, 0 m, 8 July 2014, Saarela, Sokoloff & Bull 3685 (CAN); Kugluk (Bloody Falls) Territorial Park, rocky valley immediately SW of Bloody Falls, along rough marked section of Portage Trail, upper pond just W of Bloody Falls, 67°44′39.5″N, 115°22′28.9″W ± 10 m, 15 m, 13 July 2014, Saarela, Sokoloff & Bull 3899 (CAN); Kugluk (Bloody Falls) Territorial Park, sandy NE-facing slope above small creek in deep gully, about 0.5 km W of Bloody Falls, 67°44′36.6″N, 115°22′59.3″W ± 41 m, 41 m, 15 July 2014, Saarela, Sokoloff & Bull 4019 (CAN, UBC); Kugluk (Bloody Falls) Territorial Park, top of sandy ridge, ca. 0.75 km W of Bloody Falls., 67°44′45.7″N, 115°23′4.6″W ± 25 m, 56 m, 15 July 2014, Saarela, Sokoloff & Bull 4043 (CAN); SW-facing slope above Bloody Falls, SE side of Coppermine River, 67°44′27.2″N, 115°22′58″W ± 5 m, 68 m, 19 July 2014, Saarela, Sokoloff & Bull 4197 (CAN, US).

Salix niphoclada × Salix glauca

Two of our collections correspond to this hybrid, both from Kugluk (Bloody Falls) Territorial Park. The hybrid is newly recorded for the study area.

Specimens Examined: Canada. Nunavut: Kitikmeot Region: Kugluk (Bloody Falls) Territorial Park, S-facing cliff (gabbro sill) above start of Bloody Falls, W side of Coppermine River, W side of Portage Trail, 67°44′23.2″N, 115°22′54.5″W ± 50 m, 57 m, 16 July 2014, Saarela, Sokoloff & Bull 4084 (CAN); Kugluk (Bloody Falls) Territorial Park, W side of Coppermine River, between Sandy Hills and Bloody Falls, 67°45′10.6″N, 115°22′13.1″W ± 3 m, 17 m, 17 July 2014, Saarela, Sokoloff & Bull 4142 (CAN, UBC).

Salix ovalifolia var. arctolitoralis (Hultén) Argus—Arctic seashore willow | American Beringian | Noteworthy Record

First record for Nunavut, and an eastern range extension from the Mackenzie Delta area, the previously known eastern limit (Argus, 1969, 1973; Porsild & Cody, 1980; Argus, 2004). The same limit is recorded in Argus (2007) for Salix ovalifolia s.l. (limits of infraspecific taxa not recorded there). We made two collections, one along the rocky shore of Tundra Lake just below a steep southeast-facing slope, the other in Kugluktuk in low shrub tundra amongst rocky outcrops.

Specimens Examined: Canada. Nunavut: Kitikmeot Region: W shore of Tundra Lake, ca. 4.3 km SE of Sandstone Rapids, Coppermine River, 67°25′39.2″N, 115°33′11.5″W ± 5 m, 252 m, 5 July 2014, Saarela, Sokoloff & Bull 3441 (CAN, UBC); W of Kugluktuk on tundra flats above Coppermine River, S of 1 Coronation Drive and N of power plant, 67°49′28.97″N, 115°5′0.2″W ± 100 m, 8 m, 25 July 2014, Saarela, Sokoloff & Bull 4384 (ALA, ALTA, CAN).

Salix ovalifolia Trautv. var. ovalifolia—Oval-leaved willow | Amphi-Beringian | Noteworthy Record

Newly recorded for Nunavut, and our collection represents an eastern range extension from the nearest known sites in northern Yukon (Argus, 1969, 1973; Cody, 2000). We are not aware of collections from Northwest Territories. The taxon was growing near Heart Lake in wet, hummocky tundra around small ponds with Andromeda polifolia, Betula glandulosa, Carex membranacea and Rubus chamaemorus. Argus (2010) recorded Salix ovalifolia s.l. from Northwest Territories, Yukon, Alaska and Asia (Chukotka, Russia). In addition to var. arctolitoralis (see above), two other varieties are recorded for North America, var. glacialis (Andersson) Argus from Arctic Alaska and var. cyclophylla (Rydb.) C. R. Ball from Alaska and Asia (Chukotka, Russia) (Argus, 2010; Elven et al., 2011).

Specimens Examined: Canada. Nunavut: Kitikmeot Region: ca. 0.5 km SW of Heart Lake, SW of Kugluktuk, 7.5 km SW of mouth of Coppermine River, 67°47′52″N, 115°14′14.4″W ± 350 m, 66 m, 23 July 2014, Saarela, Sokoloff & Bull 4285 (CAN, UBC).

Salix planifolia Pursh—Tea-leaved willow | North American (N) | Noteworthy Record

Newly recorded for the study area and our collections, from Fockler Creek, Kugluk (Bloody Falls) Territorial Park and near Heart Lake, represent a northern range extension with respect to the map in Argus (2007). The nearest records are from eastern Great Bear Lake (Porsild & Cody, 1980) and Bathurst Inlet (Cody, Scotter & Zoltai, 1984). Elsewhere in the Canadian Arctic recorded from a few mainland Nunavut sites, Baffin Island, islands in Hudson Bay, and northern Quebec and Labrador (Porsild & Cody, 1980; Cody, Scotter & Zoltai, 1984; Korol, 1992; Cody & Reading, 2005; Aiken et al., 2007; Argus, 2015; Bennett, 2015).

Specimens Examined: Canada. Nunavut: Kitikmeot Region: flats on W side of Fockler Creek, above spruce forest in creek valley, ca. 2.2 km S of Sandstone Rapids, Coppermine River, 67°25′49″N, 115°37′55″W ± 50 m, 152 m, 1 July 2014, Saarela, Sokoloff & Bull 3124 (CAN, UBC); Kugluk (Bloody Falls) Territorial Park, day-use area above Bloody Falls (at outhouse and fire pit), 67°44′36.8″N, 115°22′11.1″W ± 25 m, 28 m, 12 July 2014, Saarela, Sokoloff & Bull 3843 (ALA, CAN); Kugluk (Bloody Falls) Territorial Park, day-use area above Bloody Falls (at outhouse and fire pit), 67°44′36.8″N, 115°22′11.1″W ± 25 m, 28 m, 12 July 2014, Saarela, Sokoloff & Bull 3844 (CAN); ca. 0.5 km SW of Heart Lake, SW of Kugluktuk, 7.5 km SW of mouth of Coppermine River, 67°47′52″N, 115°14′14.4″W ± 350 m, 66 m, 23 July 2014, Saarela, Sokoloff & Bull 4287 (ALTA, CAN).

Salix pseudomyrsinites Andersson—Tall blueberry willow | North American (N) | Noteworthy Record

Our collection from along the upper shore of the Coppermine River at the southern limit of the Arctic ecozone is a major northern range extension for the species, and the first collection for mainland Nunavut. Our collection was taken from a low shrub ca. one foot tall growing in low shrub tundra. The nearest known sites are from southern Northwest Territories (Argus, 2007). Elsewhere in Nunavut known only from Akimiski Island in James Bay (Argus, 2010). This species was included in Salix myrtillifolia Andersson. in Porsild & Cody (1980).

Specimens Examined: Canada. Nunavut: Kitikmeot Region: S-facing slopes above Coppermine River and below spruce forest, ca. 7.8 km NNE of Sandstone Rapids, 67°31′16.2″N, 115°36′52.1″W ± 200 m, 110 m, 8 July 2014, Saarela, Sokoloff & Bull 3648 (CAN, UBC).

Salix pulchra Cham.—Diamond-leaved willow | European (NE)–Asian (N)–amphi-Beringian

The map in Porsild & Cody (1980) records this species as occurring across much of mainland Nunavut, and previously recorded for the study area (Cody, 1954b). Many of these mainland Nunavut specimens have been re-determined as other species. Argus mapped this taxon as occurring as far east as the Nunavut/Northwest Territories border, but not extending into Nunavut (Argus, 2007, Argus in Aiken et al., 2007), and later recorded it for Nunavut (Argus, 2010). Argus (2010) noted, under the closely-related species Salix planifolia, that specimens identified as Salix pulchra occur as far east as northeast of Bathurst Inlet. Five collections from the study area (Findlay 37, 42, 43, 46, 212) were published as Salix pulchra (Cody, 1954b), with a comment that they are probably referable to Salix pulchra var. yukonensis C. K. Schneid. In 1969 one of these collections was annotated by G. W. Argus as Salix planifolia subsp. pulchra var. yukonensis (C. K. Schneid.) Argus, and the other four as Salix planifolia subsp. pulchra (Cham.) Argus var. pulchra. These names are all now considered synonyms of Salix pulchra (Argus, 2010), as treated here. Elven et al. (2011) recognised two subspecies in Salix pulchra.

Specimens Examined: Canada. Nunavut: Kitikmeot Region: Coppermine [Kugluktuk], 67°49′36″N, 115°5′36″W, W. I. Findlay 37 (DAO-32631 01-01000684855); Coppermine [Kugluktuk], 67°49′36″N, 115°5′36″W, W. I. Findlay 42 (DAO-32632 01-01000684856); Coppermine [Kugluktuk], 67°49′36″N, 115°5′36″W, W. I. Findlay 43 (DAO-32633 01-01000684853); Coppermine [Kugluktuk], 67°49′36″N, 115°5′36″W, W. I. Findlay 46 (DAO-32634 01-01000684854); Coppermine [Kugluktuk], 67°49′36″N, 115°5′36″W, W. I. Findlay 212 (DAO-32639 01-01000684857); forest and slopes at confluence of Big Creek and Coppermine River, N side of Coppermine River, S side of Coppermine Mountains, 67°14′29.3″N, 116°2′44.5″W ± 250 m, 180–199 m, 7 7 July 2014, Saarela, Sokoloff & Bull 3548 (ALA, CAN); Kugluk (Bloody Falls) Territorial Park, day-use area above Bloody Falls (at outhouse and fire pit), 67°44′36.8″N, 115°22′11.1″W ± 25 m, 28 m, 20 July 2014, Saarela, Sokoloff & Bull 4233 (CAN).

Salix reticulata L., Fig. S46—Net-vein willow | Circumpolar–alpine

Previously recorded from Kugluktuk (Porsild & Cody, 1980). We made collections at Fockler Creek and Kugluk (Bloody Falls) Territorial Park. Widely distributed across the Canadian Arctic, excluding most of the high Arctic islands (Porsild & Cody, 1980; Cody, Scotter & Zoltai, 1989; Korol, 1992; Cody & Reading, 2005; Aiken et al., 2007; Argus, 2007; Saarela et al., 2013a; Argus, 2015).

Specimens Examined: Canada. Nunavut: Kitikmeot Region: Coppermine [Kugluktuk] [67°49′36″N, 115°5′36″W ± 1.5 km], J. A. Larsen s.n. (CAN-353570); Coppermine [Kugluktuk], 67°49′36″N, 115°5′36″W, 29 June 1951, W. I. Findlay 50 (DAO-185810 01-01000677363) & W. I. Findlay 51 (DAO-185811 01-01000677812); Coppermine [Kugluktuk], 67°49′36″N, 115°5′36″W, 20 June 1951, W. I. Findlay 26 (DAO-185812 01-01000677362); Coppermine [Kugluktuk], 67°49′36″N, 115°5′36″W, 2 August 1951, W. I. Findlay 237 (ACAD-30944, DAO-185809 01-01000677813); Kugluktuk, tundra slope overlooking Coppermine River [67°49′36″N, 115°5′36″W], 14 June 2000, L. K. Benjamin s.n. (ACAD-ECS015883); Kugluktuk, rocky slopes of North Hill, 67°49′29.6″N, 115°6′31″W ± 50 m, 50 m, 29 June 2014, Saarela, Sokoloff & Bull 3046 (CAN, UBC); old riverbed of Fockler Creek, ca. 2.3 km SSE of Sandstone Rapids, Coppermine River, 67°25′45.7″N, 115°37′21.8″W ± 25 m, 166 m, 1 July 2014, Saarela, Sokoloff & Bull 3166 (ALA, CAN); Kugluk (Bloody Falls) Territorial Park, rocky valley immediately SW of Bloody Falls, along rough marked section of Portage Trail, 67°44′34″N, 115°22′16″W ± 50 m, 20 m, 13 July 2014, Saarela, Sokoloff & Bull 3863 (CAN); Kugluk (Bloody Falls) Territorial Park, rocky valley immediately SW of Bloody Falls, along rough marked section of Portage Trail, 67°44′34″N, 115°22′16″W ± 50 m, 20 m, 13 July 2014, Saarela, Sokoloff & Bull 3864 (CAN).

Salix richardsonii Hook., Fig. 78—Richardson’s willow | Asian (N)–amphi-Beringian–North American (NW)

Figure 78 Salix richardsonii.

(A) Staminate catkins, Saarela et al. 3885. (B) Pistillate catkin, Saarela et al. 3885. (C) Habit, Saarela et al. 3885. Photographs by R. D. Bull (A, C) and J. M. Saarela (B).

Previously recorded from Kugluktuk (Macoun & Holm, 1921; Cody, 1954b; Porsild & Cody, 1980, as Salix lanata subsp. richardsonii (Hook.) A. K. Skvortsov). We made collections at Kugluktuk and Kugluk (Bloody Falls) Territorial Park. Elsewhere in the Canadian Arctic recorded from Baffin, Banks, Prince of Wales, Southampton and Victoria islands, and mainland Nunavut and Northwest Territories (Porsild & Cody, 1980; Cody, Scotter & Zoltai, 1984; Korol, 1992; Cody & Reading, 2005; Aiken et al., 2007; Argus, 2007; Saarela et al., 2013a).

Specimens Examined: Canada. Nunavut: Kitikmeot Region: Coppermine [Kugluktuk], vicinity of post [67°49′36″N, 115°5′36″W ± 1.5 km], 26 July 1949, A. E. Porsild 17169 (CAN-127661); Coppermine [Kugluktuk] [67°49′36″N, 115°5′36″W ± 1.5 km], 1962, J. A. Larsen s.n. (CAN-352277); Coppermine [Kugluktuk], 67°49′36″N, 115°5′36″W, 15 June 1951, W. I. Findlay 8 (DAO-185824 01-01000677964) & W. I. Findlay 9 (ACAD-30941, DAO-185827 01-01000677965); Coppermine [Kugluktuk], 67°49′36″N, 115°5′36″W, 7 June 1951, W. I. Findlay 5 (DAO-185829 01-01000677966) & W. I. Findlay 4 (ACAD-30943, DAO-185826 01-01000677503); Coppermine [Kugluktuk], 67°49′36″N, 115°5′36″W, 23 June 1951, W. I. Findlay 27 (DAO-185825 01-01000677967) & W. I. Findlay 28 (DAO-185828 01-01000677502); Kugluktuk, rocky slopes of North Hill, 67°49′29.6″N, 115°6′31″W ± 50 m, 50 m, 29 June 2014, Saarela, Sokoloff & Bull 3051 (CAN, MO, MT); Kugluktuk, rocky slopes of North Hill, 67°49′29.6″N, 115°6′31″W ± 50 m, 29 June 2014, Saarela, Sokoloff & Bull 3052 (CAN); NW-facing slope above tributary of Fockler Creek, ca. 2.4 km SSW of Sandstone Rapids, Coppermine River, 67°25′46″N, 115°38′49.4″W ± 50 m, 149 m, 3 July 2014, Saarela, Sokoloff & Bull 3297 (CAN, UBC); Kugluk (Bloody Falls) Territorial Park, slope above Bloody Falls (W side) just below Coppermine River narrows to falls, between Portage Trail and river, 67°44′25.4″N, 115°22′31.2″W ± 25 m, 29 m, 12 July 2014, Saarela, Sokoloff & Bull 3851 (ALA, CAN); Kugluk (Bloody Falls) Territorial Park, rocky valley immediately SW of Bloody Falls, along rough marked section of Portage Trail, 67°44′34″N, 115°22′16″W ± 50 m, 20 m, 13 July 2014, Saarela, Sokoloff & Bull 3885 (ALTA, CAN); Kugluk (Bloody Falls) Territorial Park, gentle stream in shallow valley running into Coppermine River just W of Bloody Falls, 67°44′36.6″N, 115°22′59.3″W ± 20 m, 41 m, 15 July 2014, Saarela, Sokoloff & Bull 4014 (CAN); W of Kugluktuk on tundra flats above Coppermine River, S of 1 Coronation Drive and N of power plant, 67°49′28.97″N, 115°5′0.2″W ± 100 m, 8 m, 25 July 2014, Saarela, Sokoloff & Bull 4380 (CAN, O).

Saxifragaceae [3/10]

Chrysosplenium rosendahlii Packer, Fig. 79A—Rosendahl’s golden-saxifrage | Amphi-Beringian–North American (N) | Noteworthy Record

Figure 79 Chrysosplenium rosendahlii and Micranthes foliolosa.

Chrysosplenium rosendahlii: (A) inflorescence and fruits, Saarela et al. 4310. Micranthes foliolosa: (B) habit, Saarela et al. 4382. Photographs by P. C. Sokoloff (A) and R. D. Bull (B).

Newly recorded for the study area. An early collection of this species from Kugluktuk was previously recognised as Chrysosplenium tetrandrum Th. Fr. (Cody, 1954b; Porsild & Cody, 1980). We made collections at Fockler Creek, Kugluk (Bloody Falls) Territorial Park, Heart Lake and Kugluktuk. This species was described from Somerset Island, Nunavut (Packer, 1963), later reduced to synonymy (Scoggan, 1978–1979; Aiken et al., 2007) or ignored (Porsild & Cody, 1980), and more recently recognised again as a distinct species (Freeman & Levsen, 2007; Elven et al., 2011), a status supported by plastid data (Saarela et al., 2013b). Packer (1963) cited and mapped collections only from Boothia, Adelaide and Melville peninsulas on mainland Nunavut. All previous collections from the study area are Chrysosplenium rosendahllii. A recent collection from just outside the study area, however, is confirmed as Chrysosplenium tetrandrum (Reading 456, DAO; Cody, Reading & Line, 2003), while one from the nearby Big Bend area of the Coppermine River is Chrysosplenium rosendahllii (Reading 22, DAO).

Specimens Examined: Canada. Nunavut: Kitikmeot Region: Coppermine [Kugluktuk], 67°49′36″N, 115°5′36″W, 2 July 1951, W. I. Findlay 62 (DAO-3759 01-01000619971); Coppermine [Kugluktuk], 67°49′36″N, 115°5′36″W, 8 July 1951, W. I. Findlay 105 (DAO-3760 01-01000619972); Coppermine [Kugluktuk], between R.C. Mission and DOT [67°49′36″N, 115°5′36″W ± 1.5 km], 7 7 July 1958, R. D. Wood s.n. (CAN-265527); NW-facing slope just upstream of small tributary from its confluence with Fockler Creek, ca. 2.4 km SSW of Sandstone Rapids, Coppermine River, 67°25′46″N, 115°38′49.4″W ± 200 m, 149 m, 3 July 2014, Saarela, Sokoloff & Bull 3308 (CAN); NW-facing slope above tributary of Fockler Creek, ca. 2.4 km SSW of Sandstone Rapids, Coppermine River, 67°25′46″N, 115°38′49.4″W ± 50 m, 149 m, 3 July 2014, Saarela, Sokoloff & Bull 3317 (CAN); Kugluk (Bloody Falls) Territorial Park, day-use area above Bloody Falls (at outhouse and fire pit), 67°44′36.8″N, 115°22′11.1″W ± 25 m, 28 m, 12 July 2014, Saarela, Sokoloff & Bull 3840 (CAN); N side of Heart Lake, below rocky cliff, SW of Kugluktuk, 5.64 km SW of mouth of Coppermine River, 67°48′33.4″N, 115°12′38.8″W ± 25 m, 31 m, 23 July 2014, Saarela, Sokoloff & Bull 4310 (CAN, UBC); SE edge of Kugluktuk, rocky cliffs overlooking Coppermine River, 67°49′9.2″N, 115°5′40.4″W ± 50 m, 28 m, 24 July 2014, Saarela, Sokoloff & Bull 4350 (CAN).

Micranthes foliolosa (R. Br.) Gornall, Fig. 79B—Leafy-stemmed saxifrage | Circumpolar

Previously recorded from Kugluktuk, as Saxifraga foliolosa R. Br (Cody, 1954b; Porsild & Cody, 1980). We made collections at Fockler Creek, along the Coppermine River and near Heart Lake. Widespread throughout the Canadian Arctic (Porsild & Cody, 1980; Cody, Scotter & Zoltai, 1989; Cody, Reading & Line, 2003; Aiken et al., 2007; Saarela et al., 2013a; Blondeau, 2015f).

Specimens Examined: Canada. Nunavut: Kitikmeot Region: Coppermine [Kugluktuk], 67°49′36″N, 115°5′36″W, 4 August 1951, W. I. Findlay 247 (DAO-25897 01-01000619980); tundra just S of Fockler Creek and N of unnamed tributary, ca. 2.2 km S of Sandstone Rapids, Coppermine River, 67°25′49″N, 115°38′8.9″W ± 3 m, 152 m, 3 July 2014, Saarela, Sokoloff & Bull 3316 (CAN); E end of small, unnamed lake on W bank of Coppermine River, ca. 8.3 km NNE of Sandstone Rapids, 67°31′30.8″N, 115°36′16.1″W ± 50 m, 126 m, 8 July 2014, Saarela, Sokoloff & Bull 3660 (CAN); ca. 0.5 km SW of Heart Lake, SW of Kugluktuk, 7.5 km SW of mouth of Coppermine River, 67°47′52″N, 115°14′14.4″W ± 350 m, 66 m, 23 July 2014, Saarela, Sokoloff & Bull 4282 (CAN, MT); creek just N of sewage retention pond (used as sewage outlet), 5.1 km SW of Coppermine River, 67°48′59.1″N, 115°12′5.8″W ± 25 m, 34 m, 23 July 2014, Saarela, Sokoloff & Bull 4335 (CAN, UBC).

Micranthes nivalis (L.) Small—Snow saxifrage | Circumpolar–alpine

Previously recorded from Kugluktuk, as Saxifraga nivalis L. (Cody, 1954b; Porsild & Cody, 1980). We made collections at Kugluktuk, Fockler Creek and Kugluk (Bloody Falls) Territorial Park. Widespread throughout the Canadian Arctic (Porsild & Cody, 1980; Cody, Scotter & Zoltai, 1989; Cody, Reading & Line, 2003; Aiken et al., 2007; Saarela et al., 2013a; Blondeau, 2015f).

Specimens Examined: Canada. Nunavut: Kitikmeot Region: Coppermine [Kugluktuk], 67°49′36″N, 115°5′36″W, 17 July 1951, W. I. Findlay 126 (DAO-185686 01-01000619936); Coppermine [Kugluktuk], basalt ridge south of settlement [67.816375°N, 115.1002722°W ± 350 m], 30 June 1958, R. D. Wood s.n. (CAN-265518); Coppermine [Kugluktuk], basalt ridge south of settlement [67.816375°N, 115.1002722°W ± 350 m], 30 June 1958, R. D. Wood s.n. (CAN-265518); Kugluktuk, rocky slopes of North Hill, 67°49′29.6″N, 115°6′31″W ± 50 m, 50 m, 29 June 2014, Saarela, Sokoloff & Bull 3057 (CAN); N side of Fockler Creek, ca. 1.9 km S of Sandstone Rapids, Coppermine River, 67°25′57.89″N, 115°38′3.9″W ± 10 m, 162 m, 4 July 2014, Saarela, Sokoloff & Bull 3321 (CAN); E side of Fockler Creek, ridge above creek valley before its confluence with Coppermine River, ca. 1.8 km S of Sandstone Rapids, 67°26′3.9″N, 115°38′20.4″W ± 25 m, 168 m, 4 July 2014, Saarela, Sokoloff & Bull 3337 (CAN); Kugluk (Bloody Falls) Territorial Park, upper ledges of rocky (gabbro) S-facing cliffs above the start of Bloody Falls (W bank of River), just E of Portage Trail, 67°44′21.7″N, 115°22′42.2″W ± 25 m, 46 m, 14 July 2014, Saarela, Sokoloff & Bull 3936 (CAN); Kugluk (Bloody Falls) Territorial Park, flats on top of mountain on W side of Coppermine River, just S of the start of Bloody Falls Rapids, 67°44′2.8″N, 115°23′39.3″W ± 250 m, 110 m, 14 July 2014, Saarela, Sokoloff & Bull 3995 (CAN).

Micranthes porsildiana (Calder & Savile) Elven & D. F. Murray, Fig. 80—Porsild’s saxifrage | Amphi-Beringian (E) | Noteworthy Record

Figure 80 Micranthes porsildiana.

(A) Inflorescences, Saarela et al. 4251. (B) habit, Saarela et al. 4251. Photographs by P. C. Sokoloff.

Our collection, and one from 1958 (Wood s.n.) not mapped in Calder & Savile (1960) or Porsild & Cody (1980), are the first records for the study area. Our collection was made in Kugluktuk, in a grassy meadow along an ATV trail growing with Calamagrostis spp., Chamerion angustifolium, Leymus mollis subsp. villosissimus and Senecio lugens. This species was recently collected from three sites just outside the study area (Reading 24-1, 81, 519, DAO; Cody, Reading & Line, 2003). These collections close a distribution gap between the Bathurst Inlet area, Hood River and the eastern shore of Great Bear Lake (Calder & Savile, 1960; Gould & Walker, 1997; Elven et al., 2011). Elsewhere in the Canadian Arctic known only from sites on central mainland Nunavut (Porsild & Cody, 1980; Cody & Reading, 2005). Porsild & Cody (1980) (after Calder & Savile, 1960) recognised this taxon as Saxifraga punctata subsp. porsildiana Calder & Taylor; however, the name Saxifraga punctata L. s.s. (syn. Micranthes punctata (L.) Losinsk.) has been misapplied in North America and that taxon is restricted to the Russian Far East (Elven et al., 2011). Brouillet & Elvander (2009a) recognised it as Micranthes nelsoniana var. porsildiana (Calder & Savile) Gornall & H. Ohba, while Elven et al. (2011) recognised it as a separate species. We follow the latter taxonomy.

Specimens Examined: Canada. Nunavut: Kitikmeot Region: Coppermine [Kugluktuk] [67°49′36″N, 115°5′36″W ± 1.5 km], 5 July 1958, R. D. Wood s.n. (CAN-265517); W of Kugluktuk on tundra flats above Coppermine River, S of 1 Coronation Drive and N of community power plant, 67°49′28.97″N, 115°5′0.2″W ± 100 m, 8 m, 22 July 2014, Saarela, Sokoloff & Bull 4251 (CAN, UBC).

Saxifraga aizoides L., Fig. S47—Yellow mountain saxifrage | North American (N)–amphi-Atlantic–European

Previously recorded from Kugluktuk (Cody, 1954b; Porsild & Cody, 1980). We made collections at Fockler Creek, Kugluk (Bloody Falls) Territorial Park and near Heart Lake cemetery. Elsewhere in the Canadian Arctic recorded from Baffin, Banks, Coats, Ellesmere, Prince Patrick, Southampton and Victoria islands, and across the mainland (Porsild & Cody, 1980; Korol, 1992; Aiken et al., 2007; Saarela et al., 2013a; Blondeau, 2015f).

Specimens Examined: Canada. Nunavut: Kitikmeot Region: Coppermine [Kugluktuk], 67°49′36″N, 115°5′36″W, 6 August 1951, W. I. Findlay 257 (DAO-185575 01-01000620002); S of Fockler Creek, along small tributary that runs into Fockler Creek, ca. 2.3 km S of Sandstone Rapids, Coppermine River, 67°25′44.9″N, 115°38′25.9″W ± 100 m, 152 m, 3 July 2014, Saarela, Sokoloff & Bull 3266 (CAN, UBC); slopes on E side of Coppermine River, N of its confluence with Fockler Creek, ca. 0.8 km SW of Sandstone Rapids, 67°26′36.9″N, 115°38′50.1″W ± 50 m, 128 m, 4 July 2014, Saarela, Sokoloff & Bull 3393 (CAN); Kugluk (Bloody Falls) Territorial Park, rocky valley immediately SW of Bloody Falls, along rough marked section of Portage Trail, upper pond just W of Bloody Falls, 67°44′39.5″N, 115°22′28.9″W ± 10 m, 15 m, 13 July 2014, Saarela, Sokoloff & Bull 3894 (ALA, CAN); hummocky tundra just SW of sewage retaining pond, N side of road to Heart Lake cemetery, 5.4 km SW of mouth of Coppermine River, 67°48′44.6″N, 115°12′27.2″W ± 3 m, 35 m, 23 July 2014, Saarela, Sokoloff & Bull 4326 (ALTA, CAN).

Saxifraga cernua L., Fig. S48—Nodding saxifrage, bulblet saxifrage | Circumpolar–alpine

Previously recorded from Kugluktuk (Cody, 1954b; Porsild & Cody, 1980). We made collections at Fockler Creek, Coppermine Mountains, Kugluk (Bloody Falls) Territorial Park and Kugluktuk. Widespread throughout the Canadian Arctic (Porsild & Cody, 1980; Cody, Scotter & Zoltai, 1989; Korol, 1992; Cody, Reading & Line, 2003; Aiken et al., 2007; Saarela et al., 2013a; Blondeau, 2015f).

Specimens Examined: Canada. Nunavut: Kitikmeot Region: Coppermine River, Fort Hearne–Bloody Falls [67.7761972°N, 115.2037222°W ± 7.5 km], 1931, A. M. Berry 9 (CAN-65749); Coppermine [Kugluktuk], 67°49′36″N, 115°5′36″W, 24 July 1951, W. I. Findlay 180 (DAO-185618 01-01000619982, UBC-V40795); Coppermine [Kugluktuk], 67°49′36″N, 115°5′36″W, 8 July 1951, W. I. Findlay 102 (DAO-185620 01-01000619981) & W. I. Findlay 105 (DAO-185617 01-01000619928); Coppermine [Kugluktuk], 67°49′36″N, 115°5′36″W, 2 August 1995, T. Dolman 92 (LEA); Kugluktuk, airport, 21 July 2013, 67.81749°N, 115.13449°W, B. A. Bennett 13-0636 (UBC, det. B. A. Bennett, Dec. 2013); old riverbed of Fockler Creek, ca. 2.3 km SSE of Sandstone Rapids, Coppermine River, 67°25′45.7″N, 115°37′21.8″W ± 25 m, 166 m, 1 July 2014, Saarela, Sokoloff & Bull 3158 (CAN); old riverbed of Fockler Creek, ca. 2.3 km SSE of Sandstone Rapids, Coppermine River, 67°25′45.7″N, 115°37′21.8″W ± 25 m, 166 m, 2 July 2014, Saarela, Sokoloff & Bull 3175 (CAN); meadow just S of Tundra Lake, ca. 4.2 km SE of Sandstone Rapids, Coppermine River, 67°25′34.8″N, 115°33′27.8″W ± 20 m, 265 m, 5 July 2014, Saarela, Sokoloff & Bull 3439 (CAN); flats atop and upper slopes of Coppermine Mountains, N/W side of Coppermine River, 67°14′43.7″N, 115°38′51.2″W ± 150 m, 422 m, 9 July 2014, Saarela, Sokoloff & Bull 3754 (CAN, UBC); Kugluk (Bloody Falls) Territorial Park, flats on top of mountain on W side of Coppermine River, just S of the start of Bloody Falls Rapids, 67°44′2.8″N, 115°23′39.3″W ± 250 m, 110 m, 14 July 2014, Saarela, Sokoloff & Bull 3993 (ALA, CAN); Kugluk (Bloody Falls) Territorial Park, N-facing slopes of large mountain just S of start of Bloody Falls, W side of Coppermine River, 67°44′7.7″N, 115°23′30.4″W ± 15 m, 90 m, 14 July 2014, Saarela, Sokoloff & Bull 4007 (CAN); Kugluk (Bloody Falls) Territorial Park, flats above Coppermine River valley, ca. 1 km W of Bloody Falls, 67°44′31.8″N, 115°24′25.6″W ± 5 m, 97 m, 16 July 2014, Saarela, Sokoloff & Bull 4104 (CAN, MO, MT); W of Kugluktuk on tundra flats above Coppermine River, S of 1 Coronation Drive and N of community power plant, 67°49′28.97″N, 115°5′0.2″W ± 100 m, 8 m, 22 July 2014, Saarela, Sokoloff & Bull 4236 (ALTA, CAN); W of Kugluktuk on tundra flats above Coppermine River, S of 1 Coronation Drive and N of power plant, 67°49′28.97″N, 115°5′0.2″W ± 100 m, 8 m, 25 July 2014, Saarela, Sokoloff & Bull 4363 (CAN, O).

Saxifraga hirculus L., Fig. S49—Yellow marsh saxifrage | Circumboreal-polar

Previously recorded from Kugluktuk and the mouth of the Rae River (Cody, 1954b; Porsild & Cody, 1980). We made collections at Fockler Creek, Kugluk (Bloody Falls) Territorial Park and near Heart Lake. Widespread throughout the Canadian Arctic (Porsild & Cody, 1980; Cody, Scotter & Zoltai, 1989; Korol, 1992; Cody, Reading & Line, 2003; Aiken et al., 2007; Saarela et al., 2013a; Blondeau, 2015f). Elven et al. (2011) recognised subspecies, following Hedberg (1992), with the caveat that the diagnostic characters cannot always be applied consistently. Brouillet & Elvander (2009b) did not recognise subspecies. Given the uncertainty we follow the latter treatment. In Hedberg (1992), most Canadian Arctic plants are mapped as Saxifraga hirculus subsp. propinqua (R. Br.) Á. Löve & D. Löve and a few low Arctic mainland collections are mapped as subsp. hirculus. Porsild & Cody (1980) recognised the taxon as Saxifraga hirculus var. propinqua (R. Br.) Simmons.

Specimens Examined: Canada. Nunavut: Kitikmeot Region: Coppermine River, Fort Hearne–Bloody Falls [67.7761972°N, 115.2037222°W ± 7.5 km], 1931, A. M. Berry 10 (CAN-65528); Coppermine [Kugluktuk], 67°49′36″N, 115°5′36″W, 30 July 1951, W. I. Findlay 209 (DAO-9607 01-01000619918, UBC-V40785); Coppermine [Kugluktuk], 67°49′36″N, 115°5′36″W, 10 July 1951, W. I. Findlay 108 (DAO9605 01-01000619919); Coppermine [Kugluktuk], 67°49′36″N, 115°5′36″W, 21 July 1951, W. I. Findlay 164 (DAO-9606 01-01000619917); Rae River mouth, 1 August 1955, R. E. Miller 294 (CAN-241986); Coppermine [Kugluktuk], 67°49′36″N, 115°5′36″W, 2 August 1995, T. Dolman 91 (LEA); NW-facing slope just upstream of small tributary from its confluence with Fockler Creek, ca. 2.4 km SSW of Sandstone Rapids, Coppermine River, 67°25′46″N, 115°38′49.4″W ± 200 m, 149 m, 3 July 2014, Saarela, Sokoloff & Bull 3306 (CAN); Kugluk (Bloody Falls) Territorial Park, gradual slope amongst large gullies, ca. 1 km W of Bloody Falls, 67°44′43.9″N, 115°23′20.8″W ± 3 m, 58 m, 15 July 2014, Saarela, Sokoloff & Bull 4040 (CAN); ca. 0.5 km SW of Heart Lake, SW of Kugluktuk, 7.5 km SW of mouth of Coppermine River, 67°47′52″N, 115°14′14.4″W ± 350 m, 66 m, 23 July 2014, Saarela, Sokoloff & Bull 4283 (CAN).

Saxifraga hyperborea R. Br.—Pygmy saxifrage | Circumpolar–alpine

Recorded previously from Kugluktuk (Cody, 1954b, as Saxifraga rivularis forma hyperborea (R. Br.) Hook.; Porsild & Cody, 1980). We collected it only once, at Fockler Creek. Until recently, Saxifraga rivularis L. was the name used across the Canadian Arctic (Porsild & Cody, 1980; Cody, 1996b) for what are now recognised as two species, Saxifraga rivularis and Saxifraga hyperborea (Jørgensen et al., 2006). Saxifraga rivularis comprises two subspecies, one with an amphi-Atlantic distribution, the other amphi-Beringian; neither are known from our region. Saxifraga hyperborea is more widespread, and present throughout the Canadian Arctic (Jørgensen et al., 2006; Aiken et al., 2007; Brouillet & Elvander, 2009b; Elven et al., 2011; Blondeau, 2015f). A key distinguishing the two species is given in Saarela et al. (2013a).

Specimens Examined: Canada. Nunavut: Kitikmeot Region: Coppermine [Kugluktuk], 67°49′36″N, 115°5′36″W, 10 July 1951, W. I. Findlay 107B (DAO-181558 01-01000619927); NW-facing slope above tributary of Fockler Creek, ca. 2.4 km SSW of Sandstone Rapids, Coppermine River, 67°25′46″N, 115°38′49.4″W ± 50 m, 149 m, 3 July 2014, Saarela, Sokoloff & Bull 3292 (CAN, UBC).

Saxifraga oppositifolia L. subsp. oppositifolia, Fig. S50A—Purple saxifrage | Circumpolar–alpine

Previously recorded from Kugluktuk (Cody, 1954b; Porsild & Cody, 1980). We made collections at Fockler Creek, Expeditor Cove and Kugluk (Bloody Falls) Territorial Park. Widespread throughout the Canadian Arctic (Porsild & Cody, 1980; Cody, Scotter & Zoltai, 1989; Korol, 1992; Aiken et al., 2007; Saarela et al., 2013a; Blondeau, 2015f). Taxonomy follows Elven et al. (2011) and Brouillet & Elvander (2009b). The other infraspecific taxon, subsp. smalliana (Engl. & Irmsch.) Hultén, is restricted to Alaska and Yukon (Brouillet & Elvander, 2009b). The two taxa are distinguished by a single character (sepal morphology).

Specimens Examined: Canada. Nunavut: Kitikmeot Region: Coppermine [Kugluktuk], 67°49′36″N, 115°5′36″W, 4 June 1951, W. I. Findlay 1 (DAO-185695 01-01000619908); Coppermine [Kugluktuk], 67°51′N, 115°16′W, 2 July 1972, F. Fodor N 162 (UBC-V151901); NW-facing slope above tributary of Fockler Creek, ca. 2.4 km SSW of Sandstone Rapids, Coppermine River, 67°25′46″N, 115°38′49.4″W ± 50 m, 149 m, 3 July 2014, Saarela, Sokoloff & Bull 3280 (ALA, CAN); Kugluktuk, rocky outcrop, overlooking Coppermine River [67°49′36″N, 115°5′36″W], 13 June 2000, L. K. Benjamin s.n. (ACAD-ECS015887); Coronation Gulf, NW peninsula of Expeditor Cove, ca. 9.6 km NW of Kugluktuk, 67°52′39.1″N, 115°16′43.8″W ± 10 m, 25 m, 8 July 2014, Saarela, Sokoloff & Bull 3700 (CAN); Kugluk (Bloody Falls) Territorial Park, rocky valley immediately SW of Bloody Falls, along rough marked section of Portage Trail, 67°44′34″N, 115°22′16″W ± 50 m, 20 m, 13 July 2014, Saarela, Sokoloff & Bull 3892 (ALTA, CAN).

Saxifraga tricuspidata Rottb., Figs. S50B and S50C—Prickly saxifrage | North American (N)

Previously recorded from Kugluktuk (Macoun & Holm, 1921; Cody, 1954b; Porsild & Cody, 1980). We made collections at Fockler Creek, Melville Creek and Kugluk (Bloody Falls) Territorial Park. Widespread throughout the Canadian Arctic (Porsild & Cody, 1980; Cody, Scotter & Zoltai, 1989; Korol, 1992; Aiken et al., 2007; Saarela et al., 2013a; Blondeau, 2015f).

Specimens Examined: Canada. Nunavut: Kitikmeot Region: mouth of Coppermine River, 14 July 1928, M. Harrington 15 (DAO-181614 01-01000620004); Coppermine [Kugluktuk], 67°49′36″N, 115°5′36″W, 22 July 1951, W. I. Findlay 170 (ACAD-30949, DAO-181616 01-01000620006, QFA0255275, UBC-V40784); Coppermine [Kugluktuk], 67°51′N, 115°16′W, 2 July 1972, F. Fodor N 138 (UBC-V151894); flats on W side of Fockler Creek, above spruce forest in creek valley, ca. 2.2 km S of Sandstone Rapids, Coppermine River, 67°25′49″N, 115°37′55″W ± 50 m, 152 m, 1 July 2014, Saarela, Sokoloff & Bull 3129 (CAN, UBC); confluence of Coppermine River and Melville Creek, just W of Coppermine Mountains, 67°15′52″N, 115°30′55.3″W ± 350 m, 178–190 m, 7 July 2014, Saarela, Sokoloff & Bull 3530 (ALA, CAN); Kugluk (Bloody Falls) Territorial Park, rocky cliffs and ledges directly above (W side) of Bloody Falls, just S of heavily used day-use/fishing area, 67°44′40.1″N, 115°22′4.9″W ± 20 m, 8 m, 12 July 2014, Saarela, Sokoloff & Bull 3808 (ALTA, CAN, O).

Appendix 1

Cyanobacteria

Nostocaceae

Nostoc commune Vaucher ex Bornet & Flahault

Specimens Examined: Canada. Nunavut: Kitikmeot Region: E side of Fockler Creek, in valley just above creek’s confluence with the Coppermine River, ca. 1.4 km SSW of Sandstone Rapids, 67°26′14.5″N, 115°38′34.8″W ± 50 m, 146 m, 4 July 2014, Saarela, Sokoloff & Bull 3356 (CAN); Kugluk (Bloody Falls) Territorial Park, upper ledges of rocky (gabbro) S-facing cliffs above the start of Bloody Falls (W bank of River), just E of Portage Trail, 67°44′21.7″N, 115°22′42.2″W ± 25 m, 46 m, 14 July 2014, Saarela, Sokoloff & Bull 3949 (CAN).

Lichens

Caliciaceae

Calicium trabinellum (Ach.) Ach

Specimens Examined: Canada. Nunavut: Kitikmeot Region: spruce forest along Fockler Creek, 67°25′45.7″N, 115°37′21.8″W ± 25 m, 166 m, 2 July 2014, associate of Saarela, Sokoloff & Bull 3200a (CAN).

Cyphelium pinicola Tibell

Specimens Examined: Canada. Nunavut: Kitikmeot Region: N side of Fockler Creek, ca. 1.9 km S of Sandstone Rapids, Coppermine River, 67°25′57.89″N, 115°38′3.9″W ± 10 m, 162 m, 4 July 2014, Saarela, Sokoloff & Bull 3328 (CAN); SW-facing slope above (N side) of Fockler Creek, ca. 3.2 km SE of Sandstone Rapids, Coppermine River, 67°25′26.2″N, 115°36′14″W ± 25 m, 193 m, 5 July 2014, Saarela, Sokoloff & Bull 3409 (CAN).

Candelariaceae

Candelariella citrina B. de Lesd.

Specimens Examined: Canada. Nunavut: Kitikmeot Region: flats atop and upper slopes of Coppermine Mountains, N/W side of Coppermine River, 67°14′49.9″N, 115°38′43.7″W ± 200 m, 467 m, 9 July 2014, Saarela, Sokoloff & Bull 3768 (CAN).

Cladoniaceae

Cladonia bellidiflora (Ach.) Schaer.

Specimens Examined: Canada. Nunavut: Kitikmeot Region: N side of Fockler Creek, ca. 1.9 km S of Sandstone Rapids, Coppermine River, 67°25′57.89″N, 115°38′3.9″W ± 10 m, 162 m, 4 July 2014, Saarela, Sokoloff & Bull 3322 (CAN).

Cladonia cf. gracilis (L.) Willd.

Specimens Examined: Canada. Nunavut: Kitikmeot Region: Kugluk (Bloody Falls) Territorial Park, N-facing slopes of large mountain just S of start of Bloody Falls, W side of Coppermine River, 67°44′7.7″N, 115°23′30.4″W ± 15 m, 90 m, 14 July 2014, Saarela, Sokoloff & Bull 4009 (CAN).

Cladonia chlorophaea (Flörke ex Sommerf.) Spreng. group

Specimens Examined: Canada. Nunavut: Kitikmeot Region: Kugluk (Bloody Falls) Territorial Park, flats above boardwalk W of Bloody Falls, 67°44′34.5″N, 115°22′27″W ± 100 m, 135 m, 11 July 2014, Saarela, Sokoloff & Bull 3790a (CAN).

Cladonia fimbriata (L.) Fr.

Specimens Examined: Canada. Nunavut: Kitikmeot Region: Kugluk (Bloody Falls) Territorial Park, flats above boardwalk W of Bloody Falls, 67°44′34.5″N, 115°22′27″W ± 100 m, 135 m, 11 July 2014, Saarela, Sokoloff & Bull 3785 (CAN).

Cladonia pleurota (Flörke) Schaer.

Specimens Examined: Canada. Nunavut: Kitikmeot Region: spruce forest along Fockler Creek, ca. 2.3 km SSE of Sandstone Rapids, Coppermine River, 67°25′45.7″N, 115°37′21.8″W ± 25 m, 166 m, 2 July 2014, Saarela, Sokoloff & Bull 3197 (CAN); E end of small, unnamed lake on W bank of Coppermine River, ca. 8.3 km NNE of Sandstone Rapids, 67°31′30.8″N, 115°36′16.1″W ± 50 m, 126 m, 8 July 2014, Saarela, Sokoloff & Bull 3693 (CAN); Kugluk (Bloody Falls) Territorial Park, flats above boardwalk W of Bloody Falls, 67°44′34.5″N, 115°22′27″W ± 100 m, 135 m, 16 July 2014, Saarela, Sokoloff & Bull 4065 (CAN).

Cladonia pocillum (Ach.) O. J. Rich.

Specimens Examined: Canada. Nunavut: Kitikmeot Region: Kugluk (Bloody Falls) Territorial Park, rocky valley immediately SW of Bloody Falls, along rough marked section of Portage Trail, 67°44′34″N, 115°22′16″W ± 50 m, 20 m, 13 July 2014, Saarela, Sokoloff & Bull 3884 (CAN); clay slopes and beach on E side of Coppermine River, just above start of Bloody Falls, 67°44′9.4″N, 115°22′41.2″W ± 15 m, 40 m, 19 July 2014, Saarela, Sokoloff & Bull 4230 (CAN).

Cladonia pyxidata (L.) Hoffm.

Specimens Examined: Canada. Nunavut: Kitikmeot Region: E side of Fockler Creek, in valley just above creek’s confluence with the Coppermine River, ca. 1.4 km SSW of Sandstone Rapids, 67°26′14.5″N, 115°38′34.8″W ± 50 m, 146 m, 4 July 2014, Saarela, Sokoloff & Bull 3344 (CAN); meadow just S of Tundra Lake, ca. 4.2 km SE of Sandstone Rapids, Coppermine River, 67°25′29.5″N, 115°33′50.4″W ± 50 m, 266 m, 5 July 2014, Saarela, Sokoloff & Bull 3462 (CAN); E end of small, unnamed lake on W bank of Coppermine River, ca. 8.3 km NNE of Sandstone Rapids, 67°31′30.8″N, 115°36′16.1″W ± 50 m, 126 m, 8 July 2014, Saarela, Sokoloff & Bull 3653 (CAN); Kugluk (Bloody Falls) Territorial Park, N-facing slopes of large mountain just S of start of Bloody Falls, W side of Coppermine River, 67°44′7.7″N, 115°23′30.4″W ± 15 m, 90 m, 14 July 2014, Saarela, Sokoloff & Bull 4006 (CAN).

Cladonia stellaris (Opiz) Pouzar & Vězda

Specimens Examined: Canada. Nunavut: Kitikmeot Region: NW-facing slope above tributary of Fockler Creek, ca. 2.4 km SSW of Sandstone Rapids, Coppermine River, 67°25′46″N, 115°38′49.4″W ± 50 m, 149 m, 3 July 2014, Saarela, Sokoloff & Bull 3283 (CAN); Kugluk (Bloody Falls) Territorial Park, upper ledges of rocky (gabbro) S-facing cliffs above the start of Bloody Falls (W bank of River), just E of Portage Trail, 67°44′21.7″N, 115°22′42.2″W ± 25 m, 46 m, 14 July 2014, Saarela, Sokoloff & Bull 3954 (CAN).

Hymeneliaceae

Tremolecia atrata (Ach.) Hertel

Specimens Examined: Canada. Nunavut: Kitikmeot Region: E side of Fockler Creek, in valley just above creek’s confluence with the Coppermine River, ca. 1.4 km SSW of Sandstone Rapids, 67°26′14.5″N, 115°38′34.8″W ± 50 m, 146 m, 4 July 2014, Saarela, Sokoloff & Bull 3347 (CAN).

Icmadophilaceae

Thamnolia subuliformis (Ehrh.) W.L. Culb.

Specimens Examined: Canada. Nunavut: Kitikmeot Region: Kugluktuk, flat mesa at top of North Hill, 67°49′32″N, 115°6′39″W ± 100 m, 50 m, 29 June 2014, Saarela, Sokoloff & Bull 3084a (CAN); Kugluk (Bloody Falls) Territorial Park, flats above boardwalk W of Bloody Falls, 67°44′34.5″N, 115°22′27″W ± 100 m, 135 m, 11 July 2014, Saarela, Sokoloff & Bull 3787 (CAN).

Lecanoraceae

Lecanorapolytropa (Hoffm.) Rabenh.

Specimens Examined: Canada. Nunavut: Kitikmeot Region: Coppermine River, confluence of Coppermine and Big Tree rivers, 66°56′23.8″N, 116°21′3.2″W ± 100 m, 265 m, 7 July 2014, Saarela, Sokoloff & Bull 3614b (CAN).

Lecanora circumborealis Brodo & Vitik.

Specimens Examined: Canada. Nunavut: Kitikmeot Region: spruce forest along Fockler Creek, 67°25′45.7″N, 115°37′21.8″W ± 25 m, 166 m, 2 July 2014, Saarela, Sokoloff & Bull 3198b (CAN); Kugluk (Bloody Falls) Territorial Park, W side of Coppermine River, between Sandy Hills and Bloody Falls, 67°45′17.6″N, 115°22′14.2″W ± 20 m, 76 m, 17 July 2014, Saarela, Sokoloff & Bull 4140b (CAN).

Lecanora epibryon (Ach.) Ach.

Specimens Examined: Canada. Nunavut: Kitikmeot Region: Kugluk (Bloody Falls) Territorial Park, flats above boardwalk W of Bloody Falls, 67°44′34.5″N, 115°22′27″W ± 100 m, 135 m, 11 July 2014, Saarela, Sokoloff & Bull 3788 (CAN).

Lecanora opiniconensis Brodo

Specimens Examined: Canada. Nunavut: Kitikmeot Region: S-facing slopes above Coppermine River and below spruce forest, ca. 7.8 km NNE of Sandstone Rapids, 67°31′16.2″N, 115°36′52.1″W ± 200 m, 110 m, 8 July 2014, Saarela, Sokoloff & Bull 3652 (CAN).

Lecanora subintricata (Nyl.) Th.Fr.

Specimens Examined: Canada. Nunavut: Kitikmeot Region: spruce forest along Fockler Creek, 67°25′45.7″N, 115°37′21.8″W ± 25 m, 166 m, 2 July 2014, Saarela, Sokoloff & Bull 3200a (CAN).

Lecanora symmicta (Ach.) Ach.

Specimens Examined: Canada. Nunavut: Kitikmeot Region: spruce forest along Fockler Creek, 67°25′45.7″N, 115°37′21.8″W ± 25 m, 166 m, 2 July 2014, Saarela, Sokoloff & Bull 3198a (CAN).

Lecidella stigmatea (Ach.) Hertel & Leuckert

Specimens Examined: Canada. Nunavut: Kitikmeot Region: dry rocky ridge top above SSW-facing slope above start of Bloody Falls, SE side of Coppermine River, 67°44′27.2″N, 115°22′58″W ± 150 m, 68 m, 19 July 2014, Saarela, Sokoloff & Bull 4201 (CAN).

Rhizoplaca chrysoleuca (Sm.) Zopf

Specimens Examined: Canada. Nunavut: Kitikmeot Region: esker on E side of Coppermine River, 0.6 km SSE of Muskox Rapids, 67°22′40″N, 115°42′38.5″W ± 50 m, 172 m, 7 July 2014, Saarela, Sokoloff & Bull 3613 (CAN); Coronation Gulf, NW peninsula of Expeditor Cove, ca. 9.5 km NW of Kugluktuk, 67°52′39.5″N, 115°16′43.8″W ± 5 m, 14 m, 8 July 2014, Saarela, Sokoloff & Bull 3729 (CAN).

Lecideaceae

Lecidea tessellata Flörke

Specimens Examined: Canada. Nunavut: Kitikmeot Region: Kugluk (Bloody Falls) Territorial Park, flats on top of mountain on W side of Coppermine River, just S of the start of Bloody Falls Rapids, 67°44′2.8″N, 115°23′39.3″W ± 250 m, 110 m, 14 July 2014, Saarela, Sokoloff & Bull 3996 (CAN).

Megasporaceae

Aspicilia cinerea (L.) Korber

Specimens Examined: Canada. Nunavut: Kitikmeot Region: ridge top N of Fockler Creek and S of Tundra Lake, ca. 3.8 km SE of Sandstone Rapids, Coppermine River, 67°25′20.4″N, 115°34′17.2″W ± 3 m, 273 m, 5 July 2014, Saarela, Sokoloff & Bull 3422 (CAN); Kugluk (Bloody Falls) Territorial Park, NE-facing slope of large hill just S of Bloody Falls, W side of Coppermine River, 67°44′6.6″N, 115°23′13.4″W ± 50 m, 40 m, 14 July 2014, Saarela, Sokoloff & Bull 3985 (CAN).

Nephromataceae

Nephroma expallidum (Nyl.) Nyl.

Specimens Examined: Canada. Nunavut: Kitikmeot Region: Kugluk (Bloody Falls) Territorial Park, rocky cliffs and ledges directly above (W side) of Bloody Falls, just S of heavily used day-use/fishing area, 67°44′40.1″N, 115°22′4.9″W ± 20 m, 8 m, 12 July 2014, Saarela, Sokoloff & Bull 3815 (CAN); SE edge of Kugluktuk, rocky cliffs overlooking Coppermine River, 67°49′9.2″N, 115°5′40.4″W ± 50 m, 28 m, 24 July 2014, Saarela, Sokoloff & Bull 4358 (CAN).

Ophioparmaceae

Ophioparma ventosa (L.) Norman

Specimens Examined: Canada. Nunavut: Kitikmeot Region: ridge top N of Fockler Creek and S of Tundra Lake, ca. 3.8 km SE of Sandstone Rapids, Coppermine River, 67°25′20.4″N, 115°34′17.2″W ± 3 m, 273 m, 5 July 2014, Saarela, Sokoloff & Bull 3421 (CAN).

Parmeliaceae

Alectoria ochroleuca (Hoffm.) A. Massal.

Specimens Examined: Canada. Nunavut: Kitikmeot Region: Kugluk (Bloody Falls) Territorial Park, W side of Coppermine River, between Sandy Hills and Bloody Falls, 67°45′10.6″N, 115°22′13.1″W ± 3 m, 17 m, 18 July 2014, Saarela, Sokoloff & Bull 4169 (CAN).

Arctoparmelia centrifuga (L.) Hale

Specimens Examined: Canada. Nunavut: Kitikmeot Region: E side of Fockler Creek, in valley just above creek’s confluence with the Coppermine River, ca. 1.4 km SSW of Sandstone Rapids, 67°26′14.5″N, 115°38′34.8″W ± 50 m, 146 m, 4 July 2014, Saarela, Sokoloff & Bull 3345 (CAN); Kugluk (Bloody Falls) Territorial Park, NE-facing slope of large hill just S of Bloody Falls, W side of Coppermine River, 67°44′6.6″N, 115°23′13.4″W ± 50 m, 40 m, 14 July 2014, Saarela, Sokoloff & Bull 3986 (CAN).

Arctoparmelia separata (Th.Fr.) Hale

Specimens Examined: Canada. Nunavut: Kitikmeot Region: forest and slopes at confluence of Big Creek and Coppermine River, N side of Coppermine River, S side of Coppermine Mountains, 67°14′29.3″N, 116°2′44.5″W ± 250 m, 180–199 m, 7 July 2014, Saarela, Sokoloff & Bull 3595 (CAN).

Asahinea chrysantha (Tuck.) W. L. Culb. & C. F. Culb.

Specimens Examined: Canada. Nunavut: Kitikmeot Region: SW-facing slope above (N side) of Fockler Creek, ca. 3.2 km SE of Sandstone Rapids, Coppermine River, 67°25′26.2″N, 115°36′14″W ± 25 m, 193 m, 5 July 2014, Saarela, Sokoloff & Bull 3410 (CAN).

Bryocaulon divergens (Ach.) Kärnefelt

Specimens Examined: Canada. Nunavut: Kitikmeot Region: top of ridge N of Fockler Creek, ca. 3.6 km SE of Sandstone Rapids, Coppermine River, 67°25′20″N, 115°35′40.1″W ± 5 m, 235 m, 5 July 2014, Saarela, Sokoloff & Bull 3417 (CAN).

Bryoria nitidula (Th.Fr.) Brodo & D. Hawksw.

Specimens Examined: Canada. Nunavut: Kitikmeot Region: W of Kugluktuk on tundra flats above Coppermine River, S of 1 Coronation Drive and N of community power plant, 67°49′28.97″N, 115°5′0.2″W ± 100 m, 8 m, 22 July 2014, Saarela, Sokoloff & Bull 4243 (CAN).

Bryoria simplicior (Vain.) Brodo & D. Hawksw.

Specimens Examined: Canada. Nunavut: Kitikmeot Region: confluence of Coppermine River and Melville Creek, just W of Coppermine Mountains, 67°15′52″N, 115°30′55.3″W ± 350 m, 178–190 m, 7 July 2014, Saarela, Sokoloff & Bull 3593 (CAN).

Cetraria islandica (L.) Ach.

Specimens Examined: Canada. Nunavut: Kitikmeot Region: N side of Fockler Creek, ca. 1.9 km S of Sandstone Rapids, Coppermine River, 67°25′57.89″N, 115°38′3.9″W ± 10 m, 162 m, 4 July 2014, Saarela, Sokoloff & Bull 3326 (CAN); Kugluk (Bloody Falls) Territorial Park, terrace above S-facing slopes above start of Bloody Falls, W side of Coppermine River, 67°44′27.2″N, 115°22′58″W ± 50 m, 68 m, 16 July 2014, Saarela, Sokoloff & Bull 4089 (CAN); Kugluk (Bloody Falls) Territorial Park, rocky valley immediately SW of Bloody Falls, along rough marked section of Portage Trail, 67°44′34″N, 115°22′16″W ± 50 m, 20 m, 18 July 2014, Saarela, Sokoloff & Bull 4155 (CAN).

Dactylina arctica (Hook. f.) Nyl.

Specimens Examined: Canada. Nunavut: Kitikmeot Region: Kugluktuk, flat mesa at top of North Hill, 67°49′32″N, 115°6′39″W ± 100 m, 50 m, 29 June 2014, Saarela, Sokoloff & Bull 3085 (CAN); confluence of Coppermine River and Melville Creek, just W of Coppermine Mountains, 67°15′52″N, 115°30′55.3″W ± 350 m, 178–190 m, 7 July 2014, Saarela, Sokoloff & Bull 3539 (CAN); Kugluk (Bloody Falls) Territorial Park, upper ledges of rocky (gabbro) S-facing cliffs above the start of Bloody Falls (W bank of River), just E of Portage Trail, 67°44′21.7″N, 115°22′42.2″W ± 25 m, 46 m, 14 July 2014, Saarela, Sokoloff & Bull 3953 (CAN).

Flavocetraria cucullata (Bellardi) Kärnefelt & A. Thell

Specimens Examined: Canada. Nunavut: Kitikmeot Region: Kugluk (Bloody Falls) Territorial Park, flats above boardwalk W of Bloody Falls, 67°44′34.5″N, 115°22′27″W ± 100 m, 135 m, 11 July 2014, Saarela, Sokoloff & Bull 3789 (CAN).

Flavocetraria nivalis (L.) Kärnefelt & Thell

Specimens Examined: Canada. Nunavut: Kitikmeot Region: Kugluktuk, rocky slopes of North Hill, 67°49′29.6″N, 115°6′31″W ± 50 m, 50 m, 29 June 2014, Saarela, Sokoloff & Bull 3065 (CAN); Kugluktuk, flat mesa at top of North Hill, 67°49′32″N, 115°6′39″W ± 100 m, 50 m, 29 June 2014, Saarela, Sokoloff & Bull 3083 (CAN); confluence of Coppermine and Big Tree rivers, 66°56′23.8″N, 116°21′3.2″W ± 100 m, 265 m, 7 July 2014, Saarela, Sokoloff & Bull 3615 (CAN); Coronation Gulf, NW peninsula of Expeditor Cove, ca. 9.5 km NW of Kugluktuk, 67°52′40.4″N, 115°16′38.3″W ± 10 m, 15 m, 8 July 2014, Saarela, Sokoloff & Bull 3707 (CAN).

Hypogymnia physodes (L.) Nyl.

Specimens Examined: Canada. Nunavut: Kitikmeot Region: confluence of Coppermine River and Melville Creek, just W of Coppermine Mountains, 67°15′52″N, 115°30′55.3″W ± 350 m, 178–190 m, 7 July 2014, Saarela, Sokoloff & Bull 3594 (CAN); forest and slopes at confluence of Big Creek and Coppermine River, N side of Coppermine River, S side of Coppermine Mountains, 67°14′29.3″N, 116°2′44.5″W ± 250 m, 180–199 m, 7 July 2014, Saarela, Sokoloff & Bull 3596 (CAN); Kugluk (Bloody Falls) Territorial Park, flats above boardwalk W of Bloody Falls, 67°44′34.5″N, 115°22′27″W ± 100 m, 135 m, 11 July 2014, Saarela, Sokoloff & Bull 3790b (CAN).

Masonhalea richardsonii (Hook.) Kärnefelt

Specimens Examined: Canada. Nunavut: Kitikmeot Region: flats on W side of Fockler Creek, above spruce forest in creek valley, ca. 2.2 km S of Sandstone Rapids, Coppermine River, 67°25′49″N, 115°37′55″W ± 50 m, 152 m, 1 July 2014, Saarela, Sokoloff & Bull 3139 (CAN); NW-facing slope above tributary of Fockler Creek, ca. 2.4 km SSW of Sandstone Rapids, Coppermine River, 67°25′46″N, 115°38′49.4″W ± 50 m, 149 m, 3 July 2014, Saarela, Sokoloff & Bull 3288 (CAN); meadow just S of Tundra Lake, ca. 4.2 km SE of Sandstone Rapids, Coppermine River, 67°25′29.5″N, 115°33′50.4″W ± 50 m, 266 m, 5 July 2014, Saarela, Sokoloff & Bull 3461 (CAN); Richardson Bay, confluence of Richardson and Rae rivers at Coronation Gulf, ca. 20 km WNW of Kugluktuk, 67°54′11.2″N, 115°32′27.4″W ± 200 m, 0 m, 8 July 2014, Saarela, Sokoloff & Bull 3692 (CAN); Kugluk (Bloody Falls) Territorial Park, NE-facing slope of large hill just S of Bloody Falls, W side of Coppermine River, 67°44′6.6″N, 115°23′13.4″W ± 50 m, 40 m, 14 July 2014, Saarela, Sokoloff & Bull 3984 (CAN); Kugluk (Bloody Falls) Territorial Park, flats above boardwalk W of Bloody Falls, 67°44′34.5″N, 115°22′27″W ± 100 m, 135 m, 16 July 2014, Saarela, Sokoloff & Bull 4064 (CAN); W of Kugluktuk on tundra flats above Coppermine River, S of 1 Coronation Drive and N of community power plant, 67°49′28.97″N, 115°5′0.2″W ± 100 m, 8 m, 22 July 2014, Saarela, Sokoloff & Bull 4244 (CAN).

Melanelia stygia (L.) Essl.

Specimens Examined: Canada. Nunavut: Kitikmeot Region: esker on E side of Coppermine River, 0.6 km SSE of Muskox Rapids, 67°22′40″N, 115°42′38.5″W ± 50 m, 172 m, 7 July 2014, Saarela, Sokoloff & Bull 3612 (CAN).

Parmeliopsis ambigua (Wulfen) Nyl.

Specimens Examined: Canada. Nunavut: Kitikmeot Region: Kugluk (Bloody Falls) Territorial Park, SE-facing slope above small stream in deep gully that runs into Coppermine River just below Bloody Falls, ca. 1 km W of Bloody Falls, 67°44′41.2″N, 115°23′34.8″W ± 50 m, 49 m, 15 July 2014, Saarela, Sokoloff & Bull 4036 (CAN).

Parmeliopsis hyperopta (Ach.) Arnold

Specimens Examined: Canada. Nunavut: Kitikmeot Region: spruce forest along Fockler Creek, 67°25′45.7″N, 115°37′21.8″W ± 25 m, 166 m, 2 July 2014, associate of Saarela, Sokoloff & Bull 3190b (CAN).

Usnea cf. lapponica Vainio

Specimens Examined: Canada. Nunavut: Kitikmeot Region: spruce forest along Fockler Creek, E side of Coppermine River, 67°25′45.7″N, 115°37′21.8″W ± 25 m, 166 m, 9 July 2014, Saarela, Sokoloff & Bull 4231 (CAN).

Vulpicida pinastri (Scop.) J.-E. Mattsson & M. J. Lai

Specimens Examined: Canada. Nunavut: Spruce forest along Fockler Creek, ca. 2.3 km SSE of Sandstone Rapids, Coppermine River, 67°25′45.7″N, 115°37′21.8″W ± 25 m, 166 m, 2 July 2014, Saarela, Sokoloff & Bull 3199 (CAN); Kitikmeot Region: forest and slopes at confluence of Big Creek and Coppermine River, N side of Coppermine River, S side of Coppermine Mountains, 67°14′29.3″N, 116°2′44.5″W ± 250 m, 180–199 m, 7 July 2014, Saarela, Sokoloff & Bull 3694 (CAN).

Vulpicida juniperinus (L.) J.-E. Mattsson & M.J. Lai

Specimens Examined: Canada. Nunavut: Kitikmeot Region: Kugluktuk, flat mesa at top of North Hill, 67°49′32″N, 115°6′39″W ± 100 m, 50 m, 29 June 2014, Saarela, Sokoloff & Bull 3086 (CAN); E side of Fockler Creek, in valley just above creek’s confluence with the Coppermine River, ca. 1.4 km SSW of Sandstone Rapids, 67°26′14.5″N, 115°38′34.8″W ± 50 m, 146 m, 4 July 2014, Saarela, Sokoloff & Bull 3346 (CAN).

Peltigeraceae

Peltigera aphthosa (L.) Willd.

Specimens Examined: Canada. Nunavut: Kitikmeot Region: NW-facing slope above tributary of Fockler Creek, ca. 2.4 km SSW of Sandstone Rapids, Coppermine River, 67°25′46″N, 115°38′49.4″W ± 50 m, 149 m, 3 July 2014, Saarela, Sokoloff & Bull 3284 (CAN); Kugluk (Bloody Falls) Territorial Park, day-use area above Bloody Falls (at outhouse and fire pit), 67°44′36.8″N, 115°22′11.1″W ± 25 m, 28 m, 12 July 2014, Saarela, Sokoloff & Bull 3845 (CAN); Kugluk (Bloody Falls) Territorial Park, flats above boardwalk W of Bloody Falls, 67°44′34.5″N, 115°22′27″W ± 100 m, 135 m, 16 July 2014, Saarela, Sokoloff & Bull 4063 (CAN).

Peltigera malacea (Ach.) Funk

Specimens Examined: Canada. Nunavut: Kitikmeot Region: Kugluk (Bloody Falls) Territorial Park, SE-facing slope above small stream in deep gully that runs into Coppermine River just below Bloody Falls, ca. 1 km W of Bloody Falls, 67°44′41.2″N, 115°23′34.8″W ± 50 m, 49 m, 15 July 2014, Saarela, Sokoloff & Bull 4039 (CAN).

Peltigera rufescens (Weiss) Humb.

Specimens Examined: Canada. Nunavut: Kitikmeot Region: S-facing slopes above Coppermine River and below spruce forest, ca. 7.8 km NNE of Sandstone Rapids, 67°31′16.2″N, 115°36′52.1″W ± 200 m, 110 m, 8 July 2014, Saarela, Sokoloff & Bull 3651 (CAN); W of Kugluktuk on tundra flats above Coppermine River, S of 1 Coronation Drive and N of community power plant, 67°49′28.97″N, 115°5′0.2″W ± 100 m, 8 m, 22 July 2014, Saarela, Sokoloff & Bull 4274 (CAN).

Solorina bispora (L.) Nyl.

Specimens Examined: Canada. Nunavut: Kitikmeot Region: Kugluktuk, rocky slopes of North Hill, 67°49′29.6″N, 115°6′31″W ± 50 m, 50 m, 29 June 2014, Saarela, Sokoloff & Bull 3063 (CAN); Kugluk (Bloody Falls) Territorial Park, rocky valley immediately SW of Bloody Falls, along rough marked section of Portage Trail, 67°44′34″N, 115°22′16″W ± 50 m, 20 m, 18 July 2014, Saarela, Sokoloff & Bull 4168 (CAN); SW-facing slopes of shallow gully in sand hills above Bloody Falls, SE side of Coppermine River across river from Kugluk (Bloody Falls) Territorial Park, 67°44′28.2″N, 115°22′3″W ± 15 m, 78 m, 19 July 2014, Saarela, Sokoloff & Bull 4185 (CAN).

Solorina saccata (L.) Ach.

Specimens Examined: Canada. Nunavut: Kitikmeot Region: confluence of Coppermine and Kendall rivers (NW side of Coppermine River, S side of Kendall River), 67°6′51.1″N, 116°8′18.3″W ± 150 m, 220 m, 7 July 2014, Saarela, Sokoloff & Bull 3578 (CAN).

Pertusariaceae

Ochrolechia upsaliensis (L.) A. Massal.

Specimens Examined: Canada. Nunavut: Kitikmeot Region: Kugluk (Bloody Falls) Territorial Park, flats above boardwalk W of Bloody Falls, 67°44′34.5″N, 115°22′27″W ± 100 m, 135 m, 11 July 2014, Saarela, Sokoloff & Bull 3786 (CAN).

Pertusaria excludens Nyl.

Specimens Examined: Canada. Nunavut: Kitikmeot Region: ridge top N of Fockler Creek and S of Tundra Lake, ca. 3.8 km SE of Sandstone Rapids, Coppermine River, 67°25′20.4″N, 115°34′17.2″W ± 3 m, 273 m, 5 July 2014, Saarela, Sokoloff & Bull 3423 (CAN).

Pertusaria panyrga (Ach.) A. Massal.

Specimens Examined: Canada. Nunavut: Kitikmeot Region: dry rocky ridge top above SSW-facing slope above start of Bloody Falls, SE side of Coppermine River, 67°44′27.2″N, 115°22′58″W ± 150 m, 68 m, 19 July 2014, Saarela, Sokoloff & Bull 4202 (CAN).

Physciaceae

Amandinea punctata (Hoffm.) Coppins & Scheid.

Specimens Examined: Canada. Nunavut: Kitikmeot Region: spruce forest along Fockler Creek, 67°25′45.7″N, 115°37′21.8″W ± 25 m, 166 m, 2 July 2014, Saarela, Sokoloff & Bull 3200b (CAN); Kugluk (Bloody Falls) Territorial Park, W side of Coppermine River, between Sandy Hills and Bloody Falls, 67°45′17.6″N, 115°22′14.2″W ± 20 m, 76 m, 17 July 2014, Saarela, Sokoloff & Bull 4140a (CAN).

Physconia muscigena (Ach.) Poelt

Specimens Examined: Canada. Nunavut: Kitikmeot Region: Kugluktuk, rocky slopes of North Hill, 67°49′29.6″N, 115°6′31″W ± 50 m, 50 m, 29 June 2014, Saarela, Sokoloff & Bull 3066 (CAN).

Rinodina mniaraea (Ach.) Körber

Specimens Examined: Canada. Nunavut: Kitikmeot Region: meadow just S of Tundra Lake, ca. 4.2 km SE of Sandstone Rapids, Coppermine River, 67°25′29.5″N, 115°33′50.4″W ± 50 m, 266 m, 5 July 2014, Saarela, Sokoloff & Bull 3462 (CAN).

Rinodina turfacea (Wahlenb.) Körber

Specimens Examined: Canada. Nunavut: Kitikmeot Region: spruce forest along Fockler Creek, ca. 2.3 km SSE of Sandstone Rapids, Coppermine River, 67°25′45.7″N, 115°37′21.8″W ± 25 m, 166 m, 2 July 2014, Saarela, Sokoloff & Bull 3190b (CAN).

Tetramelas papillatus (Sommerf.) Kalb

Specimens Examined: Canada. Nunavut: Kitikmeot Region: clay slopes and beach on E side of Coppermine River, just above start of Bloody Falls, 67°44′9.4″N, 115°22′41.2″W ± 15 m, 40 m, 19 July 2014, Saarela, Sokoloff & Bull 4228 (CAN).

Rhizocarpaceae

Rhizocarpon geographicum (L.) DC.

Specimens Examined: Canada. Nunavut: Kitikmeot Region: Kugluktuk, flat mesa at top of North Hill, 67°49′32″N, 115°6′39″W ± 100 m, 50 m, 29 June 2014, Saarela, Sokoloff & Bull 3084b (CAN); E side of Fockler Creek, ridge above creek valley before its confluence with Coppermine River, ca. 1.8 km S of Sandstone Rapids, 67°26′3.9″N, 115°38′20.4″W ± 25 m, 168 m, 4 July 2014, Saarela, Sokoloff & Bull 3343 (CAN); Kugluk (Bloody Falls) Territorial Park, upper ledges of rocky (gabbro) S-facing cliffs above the start of Bloody Falls (W bank of River), just E of Portage Trail, 67°44′21.7″N, 115°22′42.2″W ± 25 m, 46 m, 14 July 2014, Saarela, Sokoloff & Bull 3955 (CAN).

Rhizocarpon grande (Flörke ex Flotow) Arnold

Specimens Examined: Canada. Nunavut: Kitikmeot Region: Kugluk (Bloody Falls) Territorial Park, NE-facing slope of large hill just S of Bloody Falls, W side of Coppermine River, 67°44′6.6″N, 115°23′13.4″W ± 50 m, 40 m, 14 July 2014, associate of Saarela, Sokoloff & Bull 3983 (CAN).

Sphaerophoraceae

Sphaerophorus fragilis (L.) Pers.

Specimens Examined: Canada. Nunavut: Kitikmeot Region: flats atop and upper slopes of Coppermine Mountains, N/W side of Coppermine River, 67°14′49.9″N, 115°38′43.7″W ± 200 m, 467 m, 9 July 2014, Saarela, Sokoloff & Bull 3776 (CAN).

Teloschistaceae

Fulgensia cf. subbracteata (Nyl.) Poelt

Specimens Examined: Canada. Nunavut: Kitikmeot Region: SSW-facing slopes above start of Bloody Falls, SE side of Coppermine River, 67°44′12.5″N, 115°22′31″W ± 50 m, 50–60 m, 19 July 2014, Saarela, Sokoloff & Bull 4206 (CAN).

Polycauliona candelaria (L.) Frödén, Arup & Søchting

Specimens Examined: Canada. Nunavut: Kitikmeot Region: N side of Fockler Creek, ca. 1.9 km S of Sandstone Rapids, Coppermine River, 67°25′57.89″N, 115°38′3.9″W ± 10 m, 162 m, 4 July 2014, Saarela, Sokoloff & Bull 3327 (CAN).

Rusavskia elegans (Link) S. Y. Kondr. & Kärnefelt

Specimens Examined: Canada. Nunavut: Kitikmeot Region: E side of Fockler Creek, ridge above creek valley before its confluence with Coppermine River, ca. 1.8 km S of Sandstone Rapids, 67°26′3.9″N, 115°38′20.4″W ± 25 m, 168 m, 4 July 2014, Saarela, Sokoloff & Bull 3342 (CAN); Kugluk (Bloody Falls) Territorial Park, upper ledges of rocky (gabbro) S-facing cliffs above the start of Bloody Falls (W bank of River), just E of Portage Trail, 67°44′21.7″N, 115°22′42.2″W ± 25 m, 46 m, 14 July 2014, Saarela, Sokoloff & Bull 3957 (CAN); Kugluk (Bloody Falls) Territorial Park, NE-facing slope of large hill just S of Bloody Falls, W side of Coppermine River, 67°44′6.6″N, 115°23′13.4″W ± 50 m, 40 m, 14 July 2014, Saarela, Sokoloff & Bull 3983 (CAN).

Umbilicariaceae

Lasallia pensylvanica (Hoffm.) Llano

Specimens Examined: Canada. Nunavut: Kitikmeot Region: Coppermine River, confluence of Coppermine and Big Tree rivers, 66°56′23.8″N, 116°21′3.2″W ± 100 m, 265 m, 7 July 2014, Saarela, Sokoloff & Bull 3614a (CAN).

Umbilicaria lyngei Schol.

Specimens Examined: Canada. Nunavut: Kitikmeot Region: Kugluk (Bloody Falls) Territorial Park, upper ledges of rocky (gabbro) S-facing cliffs above the start of Bloody Falls (W bank of River), just E of Portage Trail, 67°44′21.7″N, 115°22′42.2″W ± 25 m, 46 m, 14 July 2014, Saarela, Sokoloff & Bull 3952 (CAN).

Umbilicaria proboscidea (L.) Schrader

Specimens Examined: Canada. Nunavut: Kitikmeot Region: top of ridge N of Fockler Creek, ca. 3.6 km SE of Sandstone Rapids, Coppermine River, 67°25′20″N, 115°35′40.1″W ± 5 m, 235 m, 5 July 2014, Saarela, Sokoloff & Bull 3460 (CAN).

Verrucariaceae

Staurothele drummondii (Tuck.) Tuck.

Specimens Examined: Canada. Nunavut: Kitikmeot Region: SSW-facing slopes above start of Bloody Falls, SE side of Coppermine River, 67°44′12.5″N, 115°22′31″W ± 50 m, 50–60 m, 19 July 2014, Saarela, Sokoloff & Bull 4205 (CAN).

Unknown

Specimens Examined: Canada. Nunavut: Kitikmeot Region: forest and slopes at confluence of Big Creek and Coppermine River, N side of Coppermine River, S side of Coppermine Mountains, 67°14′29.3″N, 116°2′44.5″W ± 250 m, 180–199 m, 7 July 2014, Saarela, Sokoloff & Bull 3597 (CAN).

Bryophyte collection numbers

Saarela et al. 3201, 3232, 3285–3287, 3356, 3399, 3425, 3590, 3591, 3663, 3689, 3698, 3731, 3749, 3753, 3791–3795, 3814, 3816, 3846, 3870–3876, 3932, 3933, 3949, 3956, 4022–4024, 4045, 4066, 4080, 4156, 4157, 4167, 4186, 4200, 4288, 4289, 4316, 4367, 4423.

Fungi collection numbers

Saarela et al. 3780, 4083, 4149, 4181–4184, 4229, 4297.

Supplemental Information

Supplemental Information 1 Histogram of the number of replicates for 2014 collections from the lower Coppermine River and vicinity.

Click here for additional data file.

Supplemental Information 2 Cystopteris fragilis and Dryopteris fragrans.

Cystopteris fragilis: (A) habit, Saarela et al. 3577. Dryopteris fragrans: (B) habitat, Saarela et al. 3434. (C) habit, Saarela et al. 3934. Photographs by R. D. Bull.

Click here for additional data file.

Supplemental Information 3 Carex marina and C. maritima.

Carex marina: (A) habit, Saarela et al. 4095. Carex maritima: (B) inflorescence, Saarela et al. 4117. (C) habit, Saarela et al. 4117. Photographs by P.C. Sokoloff.

Click here for additional data file.

Supplemental Information 4 Carex membranacea.

(A) inflorescence, Saarela et al. 3161. (B) habit, Saarela et al. 3161. Photographs by R. D. Bull.

Click here for additional data file.

Supplemental Information 5 Carex rariflora and C. saxatilis.

Carex rariflora: (A) habit, Saarela et al. 4094. Carex saxatilis: (B) habit, Saarela et al. 3607. (C) habitat, Saarela et al. 3607. Photographs by P. C. Sokoloff.

Click here for additional data file.

Supplemental Information 6 Carex scirpoidea and C. subspathacea.

Carex scirpoidea: (A) inflorescence, vicinity of Bloody Falls, Kugluk (Bloody Falls) Territorial Park, Nunavut, 13 July 2014. Carex subspathacea: (B) habit, Saarela et al. 3670. (C) habitat, Saarela et al. 3670. Photographs by R. D. Bull (A) and J. M. Saarela (B, C).

Click here for additional data file.

Supplemental Information 7 Carex vaginata.

(A) inflorescence, Saarela et al. 3919. (B) habit, Saarela et al. 3919. Photographs by P. C. Sokoloff.

Click here for additional data file.

Supplemental Information 8 Eriophorum angustifolium and E. scheuchzeri subsp. arcticum.

Eriophorum angustifolium: (A) habitat, Saarela et al. 4003. (B) habit, near Heart Lake, vicinity of Kugluktuk, Nunavut, 23 July 2014. Eriophorum scheuchzeri subsp. arcticum: (C) habit. Photographs by R. D. Bull (A, B) and P. C. Sokoloff (C).

Click here for additional data file.

Supplemental Information 9 Juncus leucochlamys.

(A) inflorescence, Saarela et al. 3854. (B) habit, Saarela et al. 3854. Photographs by J. M. Saarela.

Click here for additional data file.

Supplemental Information 10 Corallorhiza trifida.

(A) habit, Saarela et al. 4159. (B) inflorescence, Saarela et al. 3684. Photographs by R. D. Bull (A) and P. C. Sokoloff (B).

Click here for additional data file.

Supplemental Information 11 Platanthera obtusata subsp. obtusata.

(A) habit, Saarela et al. 3427. (B) inflorescence, Saarela et al. 3866. Photographs by P. C. Sokoloff.

Click here for additional data file.

Supplemental Information 12 Alopecurus borealis.

(A) inflorescence, Saarela et al. 4216. (B) habit, Saarela et al. 4216. Photographs by R. D. Bull.

Click here for additional data file.

Supplemental Information 13 Arctophila fulva.

(A) habit, Saarela et al. 4112. (B) habitat, Saarela et al. 4112. Photographs by P. C. Sokoloff.

Click here for additional data file.

Supplemental Information 14 Calamagrostis purpurascens subsp. purpurascens.

(A) inflorescence, vicinity of Fockler Creek, Nunavut, 4 July 2014. (B) habit, Kugluk (Bloody Falls) Territorial Park, Nunavut, 18 July 2014. Photographs by R. D. Bull (A) and J. M. Saarela (B).

Click here for additional data file.

Supplemental Information 15 Achillea millefolium subsp. borealis.

(A) capitulum, Saarela et al. 3959. (B) habit, Saarela et al. 3959. Photographs by R. D. Bull.

Click here for additional data file.

Supplemental Information 16 Arnica angustifolia subsp. angustifolia.

(A) capitulum, vicinity of Fockler Creek, Nunavut, 4 July 2014. (B) habit, vicinity of Tundra Lake, Nunavut, 5 July 2014. Photographs by R. D. Bull.

Click here for additional data file.

Supplemental Information 17 Arnica frigida.

(A) habit, Saarela et al.4371. (B) capitulum, Saarela et al. 4371. Photographs by R. D. Bull.

Click here for additional data file.

Supplemental Information 18 Artemisia hyperborea.

(A) habit, Saarela et al. 4020. (B) habitat, Saarela et al. 4020. Photographs by J. M. Saarela (A) and P. C. Sokoloff (B).

Click here for additional data file.

Supplemental Information 19 Hulteniella integrifolia.

(A) habit, vicinity of Fockler Creek, Nunavut, 6 July 2014. (B) capitulum, vicinity of Fockler Creek, Nunavut, 6 July 2014. Photographs by R. D. Bull.

Click here for additional data file.

Supplemental Information 20 Taraxacum ceratophorum.

(A) habit, vicinity of lower Coppermine River, Nunavut, 8 July 2014. (B) capitulum, vicinity of lower Coppermine River, Nunavut, 8 July 2014. Photographs by R. D. Bull.

Click here for additional data file.

Supplemental Information 21 Cardamine nymanii.

(A) habit, Saarela et al. 4319. (B) inflorescence, Saarela et al. 4319. Photographs by R. D. Bull.

Click here for additional data file.

Supplemental Information 22 Draba cinerea.

(A) fruits, Saarela et al. 4180. (B) habit, Saarela et al. 4199. Photographs by P. C. Sokoloff (A) and R. D. Bull (B).

Click here for additional data file.

Supplemental Information 23 Draba glabella.

(A) habit, Saarela et al. 4018. (B) fruits, Saarela et al. 4018. (C) habitat, Saarela et al. 4018. Photographs by J. M. Saarela.

Click here for additional data file.

Supplemental Information 24 Sabulina rubella.

(A) inflorescence, Saarela et al. 3096. (B) habitat, Saarela et al. 3096. Photographs by P. C. Sokoloff.

Click here for additional data file.

Supplemental Information 25 Silene acaulis and S. uralensis subsp. uralensis.

Silene acaulis: (A) inflorescences, Kugluk (Bloody Falls) Territorial Park, Nunavut, 13 July 2014. Silene uralensis subsp. uralensis: (B) inflorescence, Saarela et al. 4050. Photographs by R. D. Bull.

Click here for additional data file.

Supplemental Information 26 Andromeda polifolia and Cassiope tetragona subsp. tetragona.

Andromeda polifolia: (A) habitat, Saarela et al. 3208. (B) habit, Saarela et al. 3208. Cassiope tetragona subsp. tetragona: (C) inflorescence, vicinity of Kugluktuk Airport, 30 June 2014. Photographs by R. D. Bull (A, B) and P. C. Sokoloff (C).

Click here for additional data file.

Supplemental Information 27 Empetrum nigrum and Orthilia secunda subsp. obtusata.

Empetrum nigrum: (A) young fruits, vicinity of Kugluktuk Airport, 30 June 2014. (B) habitat, Coppermine Mountains, 9 July 2014. Orthilia secunda subsp. obtusata: (C) inflorescence, Saarela et al. 4139. (D) habit, Saarela et al. 4139. P. C. Sokoloff (A), J. M. Saarela (B) and R. D. Bull (C, D).

Click here for additional data file.

Supplemental Information 28 Pyrola grandiflora and Rhododendron lapponicum.

Pyrola grandiflora: (A) inflorescence, Kugluk (Bloody Falls) Territorial Park, Nunavut, 16 July 2014. Rhododendron lapponicum: (B) habit, Saarela et al. 3245. Photographs by R. D. Bull (A) and P. C. Sokoloff (B).

Click here for additional data file.

Supplemental Information 29 Rhododendron tomentosum subsp. decumbens and Vaccinium uliginosum.

Rhododendron tomentosum subsp. decumbens: (A) inflorescence, Kugluk (Bloody Falls) Territorial Park, Nunavut, 13 July 2014. Vaccinium uliginosum: (B) inflorescence, Kugluk (Bloody Falls) Territorial Park, Nunavut, 13 July 2014. Photographs by R. D. Bull.

Click here for additional data file.

Supplemental Information 30 Astragalus alpinus and A. richardsonii.

Astragalus alpinus: (A) habit, Saarela et al. 3159. Astragalus richardsonii: (B) habit, Kugluktuk, Nunavut, 24 July 2014. Photographs by P. C. Sokoloff (A) and R. D. Bull (B).

Click here for additional data file.

Supplemental Information 31 Oxytropis arctica and O. arctobia.

Oxytropis arctica: (A) inflorescence, Saarela et al. 3177. Oxytropis arctobia: (B) habit, Saarela et al. 4178. Photographs by P. C. Sokoloff (A) and J. M. Saarela (B).

Click here for additional data file.

Supplemental Information 32 Oxytropis deflexa subsp. foliolosa.

(A) habit, Saarela et al. 3366. (B) inflorescence, Saarela et al. 3366. Photographs by P. C. Sokoloff.

Click here for additional data file.

Supplemental Information 33 Pinguicula vulgaris subsp. vulgaris.

(A) habit, Saarela et al. 3233. (B) inflorescences, Saarela et al. 3233. Photographs by R. D. Bull.

Click here for additional data file.

Supplemental Information 34 Chamerion latifolium.

(A) inflorescence, Saarela et al. 3961. (B) habit, Saarela et al. 3961. Photographs by R. D. Bull (A) and P. C. Sokoloff (B).

Click here for additional data file.

Supplemental Information 35 Pedicularis albolabiata.

(A) habit, Saarela et al. 3230a. (B) inflorescence, Saarela et al. 3230a. Photographs by R. D. Bull.

Click here for additional data file.

Supplemental Information 36 Pedicularis labradorica and P. lanata.

Pedicularis labradorica: (A) habit, Saarela et al. 3239. Pedicularis lanata: (B) habit, Saarela et al. 3279. Photographs by P. C. Sokoloff (A) and R. D. Bull (B).

Click here for additional data file.

Supplemental Information 37 Pedicularis langsdorffii subsp. arctica.

(A) habit, Saarela et al. 3618. (B) inflorescence, Saarela et al. 3618. Photographs by P. C. Sokoloff.

Click here for additional data file.

Supplemental Information 38 Papaver hultenii.

(A) inflorescence, Saarela et al. 3451. (B) habit, Saarela et al. 3777. Photographs by J. M. Saarela (A) and R. D. Bull (B).

Click here for additional data file.

Supplemental Information 39 Parnassia kotzebuei.

(A) inflorescences, Saarela et al. 3260. (B) habit, Saarela et al. 3260. Photographs by R. D. Bull (A) and P. C. Sokoloff (B).

Click here for additional data file.

Supplemental Information 40 Armeria maritima subsp. sibirica.

(A) inflorescence, Kugluk (Bloody Falls) Territorial Park, Nunavut, 16 July 2014. (B) habit, Saarela et al. 3186. Photographs by R. D. Bull.

Click here for additional data file.

Supplemental Information 41 Androsace chamaejasme subsp. andersonii.

(A) habitat, Saarela et al. 3679. (B) habit, Saarela et al. 3679. C. inflorescences, Saarela et al. 3679. Photographs by R. D. Bull.

Click here for additional data file.

Supplemental Information 42 Primula egaliksensis.

(A) inflorescences, Saarela et al. 3234. (B) habit, Saarela et al. 3234. Photographs by P. C. Sokoloff (A) and R. D. Bull (B).

Click here for additional data file.

Supplemental Information 43 Anemone parviflora and A. richardsonii.

Anemone parviflora: (A) habit, Saarela et al. 3157. (B) habit, Kugluktuk, Nunavut, 28 June 2014. Anemone richardsonii: (C) habit, Saarela et al. 3103. (D) habitat, Saarela et al. 3103. Photographs by J. M. Saarela (A) and P. C. Sokoloff (B, C, D).

Click here for additional data file.

Supplemental Information 44 Ranunculus arcticus.

(A) habit, Saarela et al. 3706. (B) inflorescence, Saarela et al. 3706. Photographs by R. D. Bull.

Click here for additional data file.

Supplemental Information 45 Dryas integrifolia subsp. integrifolia.

(A) fruits, vicinity of Fockler Creek, Nunavut, 4 July 2014. (B) inflorescence, Kugluk (Bloody Falls) Territorial Park, Nunavut, 16 July 2014. (C) habit, vicinity of Fockler Creek, Nunavut, 2 July 2014. Photographs by R. D. Bull (A, B) and J. M. Saarela (C).

Click here for additional data file.

Supplemental Information 46 Salix reticulata.

(A) male catkin, Saarela et al. 3864. (B) female catkin, Kugluk (Bloody Falls) Territorial Park, Nunavut, 13 July 2014. (C) habit, Kugluk (Bloody Falls) Territorial Park, Nunavut, 15 July 2014. Photographs by R. D. Bull (A, B) and P. C. Sokoloff (C).

Click here for additional data file.

Supplemental Information 47 Saxifraga aizoides.

(A) habit, Saarela et al. 4326. (B) inflorescence, Saarela et al. 4326. Photographs by R. D. Bull.

Click here for additional data file.

Supplemental Information 48 Saxifraga cernua.

(A) habit, Saarela et al. 3439. (B) habitat, Saarela et al. 3439. Photographs by R. D. Bull.

Click here for additional data file.

Supplemental Information 49 Saxifraga hirculus.

(A) inflorescence, Saarela et al. 4283. (B) habitat, Saarela et al. 4283. Photographs by R. D. Bull.

Click here for additional data file.

Supplemental Information 50 Saxifraga oppositifolia and S. tricuspidata.

Saxifraga oppositifolia: (A) inflorescence, Saarela et al. 3280. Saxifraga tricuspidata: (B) inflorescence, vicinity of Fockler Creek, Nunavut, 1 July 2014. (C) habitat, Kugluk (Bloody Falls) Territorial Park, Nunavut, 14 July 2014. Photographs by P. C. Sokoloff (A, B) and R. D. Bull (C).

Click here for additional data file.

We thank the following individuals for their assistance in Kugluktuk: Rebecca (Becky) Torretti, Larry and Helen Whittaker, Nadene McMenemy, Johnny Kootoo, Grant Newman, Martin Anablak, Gerry Atatahak, Allen Kudlak Jr, and Gustin Aidjun. We thank Amanda Ward for facilitating access to the DAO collections and imaging material for this study, and Jennifer Doubt, Lyndsey Sharp, Erin Johnston and Lucie Metras for their help in the National Herbarium of Canada. We are grateful to Luc Brouillet, Paul Catling and Philip Rundel for constructive feedback on an earlier version of the manuscript.

Additional Information and Declarations

Competing Interests

Author Contributions

Field Study Permissions

DNA Deposition

Data Deposition

New Species Registration

The authors declare that they have no competing interests.

Jeffery M. Saarela conceived and designed the experiments, performed the experiments, analysed the data, wrote the paper, prepared figures and/or tables, reviewed drafts of the paper.

Paul C. Sokoloff performed the experiments, analysed the data, prepared figures and/or tables, reviewed drafts of the paper.

Roger D. Bull performed the experiments, prepared figures and/or tables, reviewed drafts of the paper.

The following information was supplied relating to field study approvals (i.e., approving body and any reference numbers):

Nunavut Department of Environment Wildlife Research Permit 2014-034;

Nunavut Territorial Parks Use Permit 2014-01;

an Inuit Owned Land Exemption Certificate from the Kitikmeot Inuit Association Department of Land, Environment and Resources to access Inuit Owned Land in the Kitikmeot Region of Nunavut;

Approval for the Use of Waters or Deposit of Waste Without a Licence (Approval number 8WLC-FCA1415) from the Nunavut Water Board.

The following information was supplied regarding the deposition of DNA sequences:

GenBank: KT906414–KT906423.

The following information was supplied regarding data availability:

The specimen citations are found throughout the annotated checklist portion of the text. New DNA sequences are available on GenBank and the accession numbers are cited in the annotated checklist.

The following information was supplied regarding the registration of a newly described species:

77158845-1—Carex petricosa subsp. misandroides

77158846-1—Carex simpliciuscula subsp. subholarctica

77158848-1—Petasites frigidus subsp. sagittatus

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
