# Peer review of "Vascular plant biodiversity of the lower Coppermine River valley and vicinity (Nunavut, Canada): an annotated checklist of an Arctic flora"

_PeerJ, doi:10.7717/peerj.2835_

## Round 0.1 · original submission · Minor Revisions

All reviewers were very positive about this paper, and so am I. Comments are minor. There is a call to reduce the length of the manuscript; however, given its scope and content - and the fact that PeerJ is an online journal, I am not convinced that this requires serious attention.

·

Basic reporting

This submission definitely sets a record for me the longest manuscript at 610 pages that I have ever been asked to review. Having said that, I applaud the authors and editor of PeerJ in working out an arrangement where a manuscript of this size with a large body of color photographs can be published. To me this is a wonderful extension of the roles that PeerJ can serve in getting open source publications available for the research community.
This is a highly detailed and scholarly treatment of the flora of the Coppermine River valley and surrounding area that is based not just on historical collections but also on a very detailed setoff new collections assembled over a month of field work in the study area. The bulk of the manuscript is the annotated checklist and associated color photographs which work well together, with the latter of high quality. The introductory and background text is also quite extensive and detailed in providing information of the environment and geography of the study area and the history of collections, followed by a detailed description of the new records and their significance. This material is all informative and well written, and he subject matter appropriate for PeerJ.
If length is an issue, the text could certainly be shortened significantly It tends to be repetitive in describing some of the facts and figures on the collections. However, for a format as in PeerJ, I don’t know that this wordiness is really important. Overall, I am very impressed by this scholarly work and would like to see it available through publication in PeerJ.

Experimental design

NA for experimental design. Rigorous field studies to complete project. Conforms to professional ethical standards.

Validity of the findings

Excellent and appropriate.

·

Basic reporting

Overall, the paper is well written and generally clear. I suggested a few minor corrections on the manuscript.
Introduction: the authors were very thorough in the introduction. No stone was left unterned. But I wonder whether this is too detailed in some ways.
Structure: The manuscript is very long and sometimes partly repetitive. It would gain in being pared down somewhat. I indicated a few areas where this could be done on the manuscript. Notably, historic aspects were partly redundant between the introduction and the history of botanizing.
Long discussions for each species were interesting but sometimes were more than the necessary. All details need not be stated in a paper. I was particularly surprized to discover the barcoding protocol for species of Botrychium embedded within a species discussion. If the protocol has been used in other papers, would it not be sufficient to simply cite these and underline modifications?
Figures: the figures associated to the text are excellent; it is nice to be able to see the landscape described and the associated vegetation.
Raw data in the form of specimens cited were supplied. However, this is increasing dramatically the length of the paper. If PeerJ had means by which these could be accessed elsewhere, it would help.

Experimental design

The work represents an investigation of a little known area of Arctic Canada; it was thorough and well substantiated. It is certainly original in terms of floristic research.
The authors justified well the nature of floristic research in their paper: it is difficult to address specific research questions in such a context. Nonetheless, the authors analyzed the data in terms of a model of arctic plant diversity, and this could have been put as a research question, with results and discussion. The paper clearly addresses knowledge gaps, however, and this seems adequate for such a paper.
The quality of the field and taxonomic work is outstanding and of high standard.
Methods were explained at great length and with much detail.

Validity of the findings

The findings presented here are novel, thorough and excellently documented.
The data are very sound and well supported.
The discussion is very long and at times includes material that would have been better summarized in table form, such as the long lists of taxa not present in the area when compared to other sites: would it be able to create one single table comparing all sites and refer to it in discussion? Although valualbe, it is incredibly tedious to have to read through long lists of names like this. And they increase the paper length considerably.
The conclusion was to the point.

·

Basic reporting

It is clear in this work that the authors are the world experts on flora of arctic Canada. The work is outstanding and sets a very high standard for floristic work of this kind that is needed in other parts of the Arctic. The introduction provides a complete and satisfactory basis. The design and layout are appropriate for floristic studies of this kind. The illustrations are of exceptionally good quality and very helpful. The submission is self-contained. All available raw data is included. The taxonomic and ecological notes are very accurate and indicate a complete knowledge of both historical and current work. New concepts, identification aids and classifications are provided, as well a new and baseline phytogeographic information. This work will be extensively used. It is well wriiten. Only few very minor corrections are required; these identified below by line number in the downloaded version:
Line Comment
404 omit near
889 spaniocarpa
1042 what about Churchill, Manitoba
1113 each of which supports
1109 was it unglaciated with some spruce at the coast
1127 omit the
1148-9 italic
1326 diminutive status or small size
1393 change an to as
1455 insert a
1549 both couplets have pinnae overlapping
2409 what corresponds to ovate or lanceolate above
3071 records
3126 from
3198 report
3340 Griseb. as below
3378 dash after QFA
2249-50 italic
4502 italic
4507 italic
4624 comma after Banks
5008-9 italic
5138 best to use nana?
5270-71 italic
5633-4 italic
5755-6 italic
5768 italic
6009-10 italic
6094 repear epithet?
6095-6 italic
6254 put from after recorded
6534 italic
6775 no subsp. here?
6880 delete (Crow)
6923 subsp. ?
7127 italic
7132 insert a
7543 insert The before subsp.
7663 send to Costea?
7912 bracket after diffusus
8078 italic
8084 Elven
8359 italic
8427 former Keewatin
8617 sites
8676 italic
8793 latter taxonomy
8882 italic
9189 italic
9458 Richardson
9530 space before PhD
9748 Keewatin
9767-68 italic
9795 Museum
9932 space before PhD
9964 space before PhD
Fig. 50 space before capillaris twice

Experimental design

This primary research fills a huge knowledge gap and this is clearly outlined. The methods are adequately described. It is rigorous and surpasses current standards for work of this kind. A high technical standard is indicated here by the use of molecular techniques to assist in taxon identification as well as extensive examination of many collections and much literature prior to decisions.

Validity of the findings

The findings are valid and supported by voucher material in a publicly accessible collection in the Canadian Museum of Nature (i.e. in a discipline-specific repository). Conclusions are appropriately stated and the information and overall conclusions are original.

Additional comments

This is not only a signicant contribution to botanical science in the areas of taxonomy, ecology and phytogeography, it is also exemplary. Increasing interest in the environment of the far north will result in this work being extensively used, and it will have substantial impact. In particular it will will set an important new standard for much work of this kind that will be needed to better understand Arctic resources.

---

## Round 0.2 · accepted · Accept

I am delighted that PeerJ is publishing your paper and hope this opens the door for more submissions like yours.